

SciPost Phys. Lect. Notes 90 (2025)

# Introduction to string theory

**Carlo Maccaferri[1★], Fabio Marino[2,3†] and Beniamino Valsesia[4‡]**

**1** Dipartimento di Fisica, Università di Torino and INFN,
Sezione di Torino, Via Pietro Giuria 1, I-10125 Torino, Italy
**2** Dipartimento di Fisica, Università di Milano-Bicocca and INFN,
Sezione di Milano-Bicocca, Piazza della Scienza 3, I-20126 Milano, Italy
**3** Department of Mathematics, University of Surrey, Guildford, GU2 7XH, UK
**4** SISSA, Via Bonomea 265, 34136 Trieste, Italy and INFN,
Sezione di Trieste, Via Valerio 2, 34127 Trieste, Italy

★ maccafer@gmail.com , † f.marino25@campus.unimib.it , ‡ bvalsesi@sissa.it

## Abstract

These are lecture notes of the introductory String Theory course held by one of the authors for the master program of Theoretical Physics at Turin University. The world-sheet approach to String Theory is pedagogically introduced in the framework of the bosonic string and of the superstring.

| | |
|---|---|
| Received | 2024-09-18 |
| Accepted | 2024-11-21 |
| Published | 2025-01-29 |

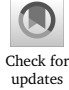

# Introduction to String Theory

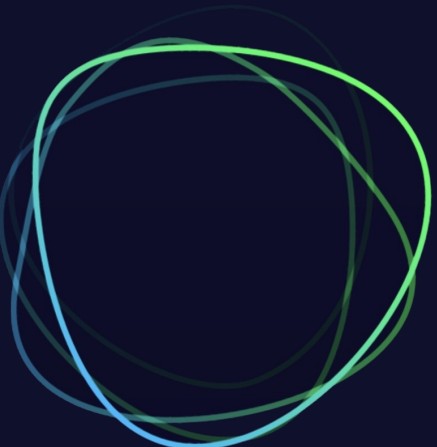

C. Maccaferri, F. Marino and B. Valsesia

University of Turin - Physics Department

*No time,*
*No space,*
*Another race of vibrations.*

Franco Battiato

## Foreword and acknowledgments

These lecture notes are not supposed to be a substitution for the textbooks and reviews that are available in the literature. In fact, they are heavily based on them. However during the course I follow a somewhat non-standard path and these notes can be useful to find the discussed material at the right place.

These notes have been originally written by F. Marino and B. Valsesia who followed the first edition of the course in 2020. The manuscript became quickly rather useful to the other students in the subsequent years. After some further editing, the notes have now reached a more-or-less stable state. That doesn't mean however that typos have been completely eliminated or that they cannot be further improved and/or enriched. We thank in advance anyone who will point us extra typos, possible issues or just suggestions.

The level of the lectures is set for undergraduate students with knowledge of basic Quantum Field Theory and General Relativity. The notes can also be appropriate for a basic PhD course on String Theory, in case of no previous exposure to the subject. No knowledge of Supersymmetry is required.

November 2023, C. Maccaferri

# 1 Introduction

## 1.1 The heritage of the past millenium

Last century has been characterized by an impressive advance in the understanding of the physical world. We have understood how Nature is organized at very small scales, following the rules of Quantum Mechanics. We have also understood how it is organized at very large scales, according to the elegant description of General Relativity. A lot of experiments, especially concerning high energy physics in accelerators, confirmed our vision of the quantum world to an astonishing precision. However more and more confusion emerged from astrophysical observations, to the point that we finally realized that what we have understood so well is only a tiny part inside a very mysterious Universe.

- The celebrated Standard Model of Particle Physics (the most precise theory we have at our disposal to describe the microscopic world) can only describe about 5% of the matter/energy which is present in the Universe. A mysterious 20% should be incarnated by the so-called *dark matter*. This is thought of as an exotic form of particle matter which is very weakly interacting with photons (and so it is invisible) but which contributes to the energy momentum tensor and thus couples to gravity. Without dark matter, galaxies would not rotate in the way we observe and the structure of the large-scale universe would be different. But what is dark matter? How is it related to the standard model matter and to the real nature of gravity? *No one knows.*

- Our universe is expanding and the expansion is accelerating. Something is constantly pushing our universe apart. How is this possible? General Relativity tells us that this is what happens when there is a positive cosmological constant in Einstein equations. Although this cosmological constant is expected to be vey small, it still provides the remaining 75% of the energy balance. Even more mysteriously than dark matter, this elusive form of energy is invoked with the name of *dark energy*. Since dark energy is at the end of the day a positive cosmological constant, it should be produced as a vacuum energy of some of the fields which describe the evolution of the universe at large scale. Which fields? How are they related to the Standard model fields? *No one knows.*

- We have finally observed black holes, also thanks to the impressive success in the detection of gravitational waves. We are essentially sure that giant black holes live at the center of every galaxy. Smaller, stellar-like black holes are everywhere in the universe. What is their physics? Are they really "black" or they can eventually give back something of what they have absorbed? Despite a lot of research and speculations, again *no one knows*.

This is empirical evidence that the Standard Model plus General Relativity cannot be the end of the story.

How can we fill the gap?

Many theoretical physicists believe that the main reason why we don't understand the above issues (and in fact many others) is that our knowledge of the basic laws of Nature is not complete and we are missing something fundamental. This belief comes from the fact that, at a fundamental theoretical level, there is a big hole in our model of the universe. Indeed it turns out that Quantum Field Theory and General Relativity, the two pillars that are at the basis of everything we know, are not compatible with one another! To understand what is the story of this incompatibility let us sketchily review the salient features of the quantum theory and of gravity.

### 1.1.1 Quantum field theory

Quantum Field Theory (QFT) is the paradigm we use to describe fundamental interactions at the microscopic scale. Most of the QFT's we are interested in have a definition in terms of a *path integral* which allows to compute correlation functions of operators $\{O_i(\phi)\}$ as

$$\langle O_1 \cdots O_n \rangle \sim \int \mathcal{D}\phi \, O_1(\phi) \cdots O_n(\phi) \, e^{\frac{i}{\hbar} S(\phi)}. \tag{1.1}$$

Assuming that the action is of the standard form

$$S(\phi) = \int d^D x \left( \frac{1}{2} \phi K \phi - V(\phi, \partial \phi) \right), \tag{1.2}$$

one is usually interested in *perturbative* computations. This means that we consider the theory of (small) fluctuations around the saddle point $\phi = 0$. The leading contribution from the saddle point is the tree-level part of the amplitude. Higher terms in the saddle point expansion account for loop corrections. This procedure gives a power series expansion in $\hbar$ which is called *perturbation theory* and it is given by the sum of all the Feynman diagrams. The obtained contributions are generically divergent but, if the original action is renormalizable (i.e. if it possible to absorb the divergences into a *finite* number of counter-terms), one finally obtains finite numbers for the scattering amplitudes, which can be tested against experiments in colliders. This is what happens for the Standard Model of particles which describes in this way the physical phenomena associated with electromagnetic, weak and strong forces. This description is very accurate when $\hbar \to 0^+$, where the quantum couplings are small and becomes however less predictive outside of this limit. The reason is that an amplitude is not in general an analytic function of $\hbar$ and the power series expansion is often only an asymptotic expansion with zero radius of convergence. This typically happens because, as $\hbar$ grows, the path integral starts to be sensible to other classical solutions $\phi_*$ of the (euclidean) action $S_E(\phi)$ which contribute at order $\sim e^{-\frac{1}{\hbar} S_E(\phi_*)}$. Notice that this quantity is evidently not vanishing, yet it has a vanishing power series expansion in $\hbar \to 0^+$. Therefore this contribution is *invisible* in perturbation theory. These effects are called *non-perturbative contributions*, the paradigmatic example being the so-called *instantons*. Although they are negligible at weak coupling, they are however important at strong coupling, for example to understand confinement of the strong interaction at low energy. It is fair to say that while we understand very well the perturbative structure of QFT (weak coupling), the non perturbative strong coupling aspects are still rather elusive. In conclusion, given the fact that we typically have weak coupling at high energy (asymptotic freedom), we understand well the UV behaviour of QFT but we still have rather poor understanding of the infrared (IR) region. However we believe (as the numerous lattice simulations have confirmed) that the path integral is giving a complete definition of QFT, we are just unable to calculate it without resorting to perturbation theory.

### 1.1.2 General relativity

The fourth fundamental force of Nature, gravity, is described by a *classical* field theory, General Relativity, encoded in the Einstein-Hilbert action

$$S_{EH}(g_{\mu\nu}) \sim \frac{1}{\kappa^2} \int d^4 x \sqrt{-g} \, (R(g) - \Lambda). \tag{1.3}$$

The quantity $\kappa \sim \sqrt{G_{\text{Newton}}}$ is the gravitational coupling and simple dimensional analysis tells us that it has dimension of length (in natural units $\hbar c = 1$), which means that gravity is expected to become strong at very short distances. This expectation is however not really testable

because at short distances we should treat gravity quantum mechanically and we cannot, because General Relativity is not renormalizable. This is expected for a theory having a dimensionful coupling constant $\kappa$ of negative mass dimension. Indeed in this case renormalization typically requires an *infinite* number of counter-terms of increasing dimension, with the consequent complete loss of predictiveness. A way out would be UV finiteness of the theory without renormalization. It turns out that pure GR (with no matter) is finite at one-loop but diverges starting from two loops. Therefore renormalization is needed and infinite counter-terms are generated in the process. Let us elaborate a bit more on why we need infinite counter-terms. Counter-terms are naturally organized in a power-series expansion in $\kappa$. Since $\kappa$ has dimension of an inverse mass, a generic counter-term will be $\sim \kappa^n \mathcal{O}_{(4+n)}$, which is an *irrelevant*[1] operator of mass dimension $\Delta = 4 + n$. Since there is no upper bound to the dimensions of irrelevant operators, this means that, to absorb the divergences, we will have to switch on infinite counter-terms allowed by general coordinate invariance. These counter-tems are higher derivatives of the curvature tensor(s) and its powers $\sim \kappa^{m+2p-4} \partial^m [R]^p$ and we would need infinite experiments to fix them. This means that the theory is non-renormalizable. As a matter of fact, this is not a peculiarity of gravity but it is common to all theories with a negative dimension coupling constant by analogous arguments.

The dimensionful coupling constant and the consequent need of infinite counter-terms is indeed reminiscent of the Fermi theory of weak interactions which is not renormalizable but it is the low-energy effective field theory of the (renormalizable) $SU(2)$ gauge theory of weak interactions, obtained by integrating out the $SU(2)$ massive gauge bosons. If we follow this analogy we are then lead to consider the possibility that GR might be an effective field theory where some fundamental microscopic degrees of freedom have been integrated out leaving the graviton as the low energy degree of freedom. As we will see, this is precisely the picture that is suggested by string theory.

To summarize, the problem with GR is in a sense opposite to the problem with QFT. In GR we understand very well the infrared (classical gravity) but we are in trouble with the ultraviolet (quantum gravity).

### 1.1.3 The Planck scale

What is the scale at which classical GR is expected to fail in describing gravity? When gravity becomes quantum mechanical? We can have a simple estimate from dimensional analysis. If we associate to Relativistic Quantum Mechanics the constants of Nature $c$ and $\hbar$ and to General Relativity $c$ and $G$, we see that we can construct a fundamental length, mass and time called *Planck units*

$$l_{\text{Planck}} = \sqrt{\frac{G\hbar}{c^3}} \sim 10^{-35} m \,, \tag{1.4}$$

$$m_{\text{Planck}} = \sqrt{\frac{\hbar c}{G}} \sim 10^{19} Gev \sim 10^{-8} Kg \,, \tag{1.5}$$

$$t_{\text{Planck}} = \sqrt{\frac{G\hbar}{c^5}} \sim 10^{-44} s \,. \tag{1.6}$$

---

[1]If a theory is defined in $D$ space-time dimensions, an irrelevant operator has mass dimension $\Delta > D$. A marginal operator has dimension $\Delta = D$ and a relevant operator has dimension $\Delta < D$. Irrelevant operators are not important at low energy but become important at high energy and, in an effective field theory understanding, are generated by integrating out high energy degrees of freedom.

The Planck mass, in particular, can be defined as the mass of a particle whose *Compton wave-lenght* $\lambda_c$ is the same as its *Schwarzschild radius* $r_s$. To be precise, remembering that

$$\lambda_c(m) = \frac{2\pi\hbar}{c}\frac{1}{m}, \tag{1.7}$$

$$r_s(m) = \frac{2G}{c^2}m, \tag{1.8}$$

the Planck mass is defined by the relation

$$r_s(m_{\text{Planck}}) = \frac{1}{\pi}\lambda_c(m_{\text{Planck}}). \tag{1.9}$$

This is essentially saying that the wavelength of a photon that can detect the particle is the same as the event horizon of the particle itself! If the mass of the particle were higher, the black hole associated to the particle would eat the photon completely. In length terms this means that it is not possible to resolve distances smaller than the Planck length $l_{\text{Planck}} \sim \lambda_c(m_{\text{Planck}})$ because the needed energy would create a black hole whose horizon is larger than the distance we want to probe! This is an important lesson that we get by trying to combine the rules of quantum mechanics with the existence of event-horizons. From QM we know that to resolve smaller and smaller distances we need higher and higher energy. However there is a limit to the energy that can be stored in a region of space: if too much mass-energy is concentrated in a too small region this will collapse gravitationally, forming a black hole! This discussion is certainly rather heuristic but it points to the fact that in any quantum theory of gravity we expect to have a minimal distance beyond which space-time itself breaks down. The existence of a minimal length also suggests that in a full quantum gravity theory there should not be UV divergences whatsoever because there is a natural cut-off given by the Planck scale. This is precisely realized in string theory, as we will see in detail.

## 1.2  String theory: A first look

The basic idea of string theory is that the fundamental degrees of freedom are extended objects: one-dimensional *relativistic* strings. The size of strings is the only scale of the theory which is called $\alpha'$ and has the dimension of length squared. The size of the string is set to be of the order of the Planck length

$$\alpha' \sim l_{\text{Planck}}^2, \tag{1.10}$$

but the precise relation is a consequence of the strings dynamics and of the string coupling constant (a quantity which will be introduced later on in the course). However we have to specify a little better what we mean by 'size': since we want a relativistic theory we cannot set a fixed length, as this would not be invariant under Lorentz transformations. What is constant is the *tension* of the string, which can be considered as the mass per unit length

$$T_{\text{string}} = \frac{1}{2\pi\alpha'}. \tag{1.11}$$

If we perform a Lorentz boost the length of the string can change but its tension remains constant (just like the invariant mass of a relativistic particle). This has the funny consequence that by stretching the string the energy grows linearly and not quadratically as it would be for a non-relativistic rubber band. Incidentally (but not that much incidentally after all) this is precisely the behaviour of QCD flux tubes between quarks in the confinement regime. This is not a curious coincidence but it is related to the fact that relativistic strings are indeed a good description of strong interactions in the strong coupling phase. This is also one of the line

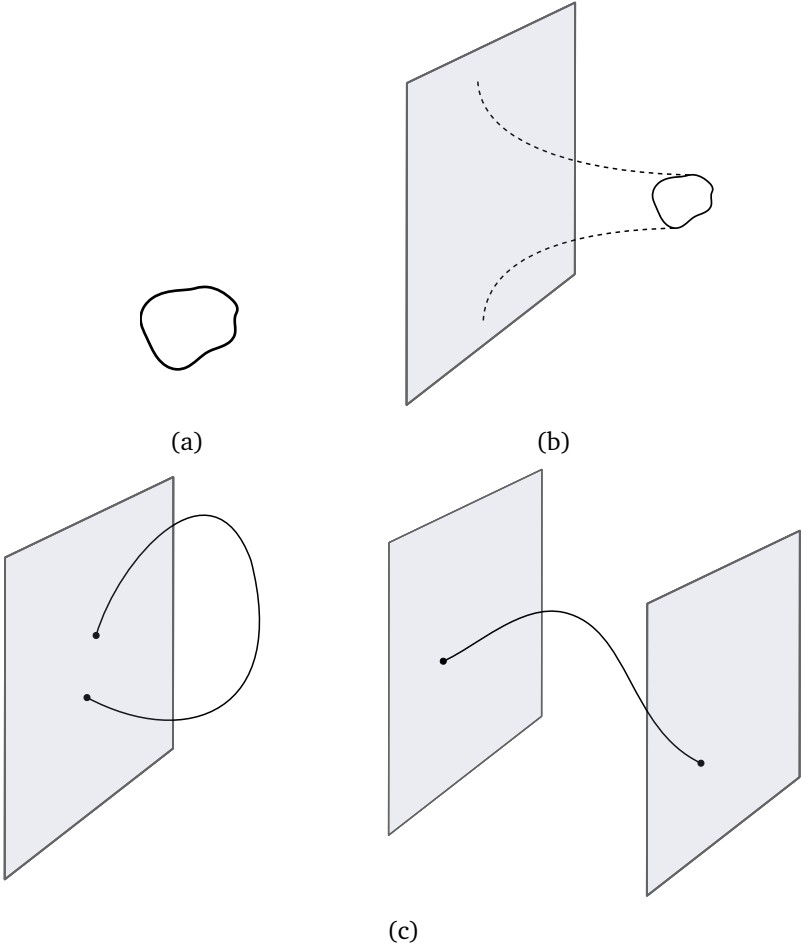

(a)        (b)

(c)

Figure 1.1: A free closed string, a closed string emitted by a *D*-brane and open strings attached to a single or multiple *D*-branes.

of reasoning that can bring to the celebrated AdS/CFT correspondence which essentially says that some QCD-like gauge theories (the 'CFT' side of the duality), when they are in a strong coupling phase, can have a dual description in terms of strings (the 'AdS' side of the duality).

Strings can close on themselves to form a loop (closed strings) or they can have endpoints (open strings). Closed strings are sort of universal objects which can freely propagate in space-time, just like particles (see fig. 1.11.1(a)). Open strings, on the other hand, need some more structure to be identified. In particular, contrary to the closed string, the endpoints of an open string are two preferred points whose position in space-time will generically be on some hypersurface (which in some case can coincide with space-time itself) (see fig. 1.11.1(c)). These surfaces are known as *D-branes* and are one of the most important ingredients of string theory. As we will see *D*-branes are genuine dynamical objects which can fluctuate and interact in space-time. In fact, the most correct way of thinking at open strings is in terms of *fluctuation modes* of *D*-branes. *D*-branes interact with closed strings. In fact *D*-branes are the sources of closed strings in the same way as usual matter is the source of the gravitational field (see fig. 1.11.1(b)).

To better understand this point we need to zoom a bit on the internal dynamics of a string. Being an extended object a string will be first of all characterized by its center of mass which will behave as an elementary particle. The mass and the spin of this elementary particle depend on the *vibrational* mode of the string. Indeed thanks to its tension the string vibrates relativistically. Vibrations naturally organize themselves into harmonics and, as we will see,

each harmonic gives rise to a state with definite mass and spin. In particular, calling $N \in \mathbb{Z}$ the vibrational mode, the mass of the state will be of the form

$$\alpha' m^2 \sim N - 1 \,, \tag{1.12}$$

where '1' is a zero-point energy whose precise value will be determined by quantum consistency and which depends on whether we are considering *bosonic strings* (as we are doing now) or *superstrings*, for which the zero point energy will be different. But these are unimportant details for the time being. At the same time, the spin of the state will also be determined by the vibrational mode in a way that roughly goes as

$$s \sim 2N \,, \qquad \text{closed string,} \tag{1.13}$$
$$s \sim N \,, \qquad \text{open string.} \tag{1.14}$$

We see that for the first harmonic $N = 1$ we get a massless particle whose spin is two for the closed string and one for the open string. Now, it can be shown that a massless spin two particle can only be the graviton: there are no consistent interactions for a spin two massless particle other than general relativity (with possibly higher derivative corrections) where the graviton $h_{\mu\nu}$ appears as the fluctuation of the metric tensor around flat Minkowski metric:

$$g_{\mu\nu} = \eta_{\mu\nu} + h_{\mu\nu}. \tag{1.15}$$

Therefore we have to conclude that a theory of closed strings necessarily includes gravity! The nice thing about this is that, as we will see in detail, strings interact consistently in a full quantum setting and since gravity is part of the closed string spectrum, string theory naturally provides a setting in which gravity is treated quantum mechanically, without encountering the problems we had in trying to quantize general relativity. String theory, beside other things, provides a theory of quantum gravity.

But gravity is not the only thing we get. Looking at the open string spectrum we also encounter massless spin-one particles. These are the needed Lorentz representations to describe the gauge bosons of Yang-Mills theories and indeed we will see that open strings (or more correctly $D$-branes) are responsible for the emergence of gauge theories from string theory. Therefore in string theory Yang-Mills theories and gravity are unified in a fully quantum mechanical way.

We have said before that in a theory of quantum gravity we expect UV divergences to be tamed by the natural cutoff provided by the Planck mass. How is this realized in string theory? It is not possible here to give a complete account of the absence of UV divergences but the following example will give you the taste of what happens in a high energy string scattering.

Consider a $\phi^3$ theory where the interaction potential is given by $V(\phi) = g \int d^D x \, \phi^3(x)$. This theory is obviously UV divergent, the simplest divergent diagram being the one-loop tadpole (see fig. 1.2) consisting of a cubic vertex where two legs are connected by a propagator and the third leg is the emitted particle from the unstable quantum vacuum

$$\mathcal{A} \sim g \int d^D p \, \frac{1}{p^2 + m^2} = \text{divergent.} \tag{1.16}$$

To get an (although incomplete) idea of what happens in string theory one should think of substituting the potential as

$$V(\phi) \to V(\tilde{\phi}), \tag{1.17}$$

where

$$\tilde{\phi} = e^{\alpha' \Box} \phi \to e^{-\alpha' p^2} \phi(p), \tag{1.18}$$

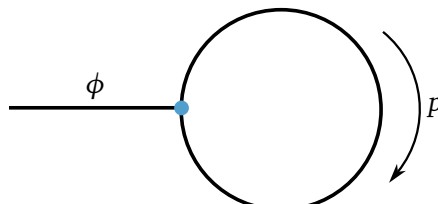

Figure 1.2: The tadpole diagram of the $\phi^3$ theory.

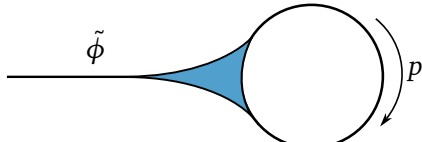

Figure 1.3: The tadpole diagram of the deformed $\tilde{\phi}^3$ theory.

and at the end we switched to momentum space. This deformation essentially dresses the interacting field with a gaussian form factor which describes a spreading of the interaction at a size $\sim \sqrt{\alpha'}$ (see fig. 1.3). This simulates the fact that strings, being extended objects, cannot interact at a point. Now the same tadpole diagram will become

$$\mathcal{A} \sim g \int d^D p \, \frac{e^{-2\alpha' p^2}}{p^2 + m^2} = \text{convergent}, \tag{1.19}$$

where we are assuming we are doing a Wick rotation to euclidean momentum to ensure convergence thanks to the gaussian factor. We see that the fact that the interaction is diffused at the string scale $\alpha'$ guarantees that UV divergences will always be absent, because the exponential suppression will always be stronger than any power coming from the loops.

During the lectures we will have a more refined understanding of how UV divergences are avoided (or reinterpreted as IR divergences), but this simple example should give the right intuition of what is happening at high energies: string theory is very soft, much softer than any standard field theory.

What we have described up to now (bosonic string theory) seems very promising for a possible ultimate theory unifying QFT and GR. There are however a couple of unsatisfactory issues which tells us that bosonic string theory is not enough.

- The theory is unstable. If you look at the mass formula (1.12), it appears that for $N = 0$ the corresponding state is a *tachyon*. In QFT a tachyon is a sign which is telling us that we are quantizing the theory around an unstable vacuum. The search for a stable vacuum is far from obvious, especially for what concerns the fate of the closed string tachyon.

- It turns out that the spins we are able to create with the bosonic string are all integer (1.13), (1.14) and therefore we can only describe spacetime bosons. But we need fermions to describe nature.

- Quantum consistency of the theory is subtle and it fixes the dimension of space-time to be 26. This is clearly very far from our 4 dimensional space-time.

As we will see in the last part of the lectures, the first two points are solved by upgrading to the so-called *superstring* which is at the moment the best available model for unification. In the superstring the dimension of space-time will be fixed to 10 by quantum consistency. This is still rather far from our 4 but, either through *compactification* or by means of various D-branes

configurations one can easily get effective 4D field theories which are qualitatively very close to the standard model. Getting *exactly* the standard model is an active area of research and, as of now, it has not yet been accomplished.

There is indeed a multitude of ways we can obtain a low energy effective field theory from a unique superstring theory in 10 D, which suggests that our "low energy" physics may be in a sense accidental, just like it is accidental that our planet is at the right distance from the sun to guarantee life. In other words, our experimented physics may be just one of the possible billions of ways string theory has broken down at low energy. This sometimes leads to speculations which inevitably have a sort of philosophical taste. This however should not distract us too much. On the other hand there is perhaps a lesson to be learnt about the difficulty of reproducing exactly the standard model which has to do with the fact that we still don't understand the real non-perturbative structure of string theory. Without this understanding it is not really possible to make solid low energy predictions. The understanding of string theory is still an on-going endeavour which will have to be continued by the next generations of physicists in the years to come!

## 1.3 Overview of the course

These lectures are rather long and extended, probably too much for a 48 hours undergraduate course. We have tried to explain everything by only assuming some knowledge of perturbative QFT (including renormalization) and a bit of GR. But this requires time and constant effort to fill the gaps. We hope we have reached the goal at least partially and students will find them useful.

The notes are organized as follows.

In chapter 2 we study the relativistic particle and develop the tools that will be later used for the string, in particular BRST quantization.

In chapter 3 we introduce the bosonic string in the oscillator formalism and we explore the spectrum of closed and open strings. In particular we introduce D-branes as boundary conditions for open strings. We go through the BRST quantization of the Polyakov path integral and we introduce the $(b, c)$ reparametrization ghosts, the BRST charge and its cohomology. At last we introduce light-cone quantization as a way to capture the gauge invariant content of BRST, at the price of loosing Lorentz covariance.

In chapter 4 we give an account of two dimensional conformal field theory, which is the worldsheet theory we get after gauge fixing the Polyakov path integral. We also give some notion of boundary conformal field theory, to properly describe open strings and D-branes. We end up describing the general bulk-boundary correlation functions and relate them to interactions between open and closed strings in presence of D-branes.

Chapter 5 deals with string perturbation theory. After discussing the topological expansion of the Polyakov path integral we discuss in some detail tree level closed and open string amplitudes, where all the CFT machinery developed in the previous chapter can be used. Then we discuss the non-linear sigma model and the emergence of Einstein equations, together with the full non-linear field equations for the massless closed string states. We continue by giving a quick look at loop amplitudes and to the concept of moduli space. As an example we discuss the closed string one-loop vacuum amplitude and we introduce modular invariance. We also discuss the open string one-loop vacuum amplitude. We carefully go through the analysis of open string UV divergences and reinterpret them as IR divergences associated to Riemann surface degeneration in the closed string channel. We then follow Polchinski original computation to see that D-branes are sources for closed strings and we estimate the tension of the D-branes.

In chapter 6 we finally introduce the superstring and its superconformal two-dimensional field theory with Neveu-Schwarz and Ramond sectors. We derive the superconformal constraints and we obtain the spectrum of physical fluctuations. By explicit computation in light-cone gauge we verify that the torus partition function is not modular invariant as it stands but needs the GSO projections. In this way, out of an inconsistent model, we get four consistent truncations: Type IIA-B and Type 0A-B. The Type 0 theories only contain spacetime bosons and are still plagued by the tachyon. On the other hand the two Type II theories are stable (no tachyons) and contains bosons and fermions. We discuss their spectrum and find out that there is a perfect match, at every mass level, between bosonic and fermionic degrees of freedom. This is the sign of spacetime *supersymmetry*. In the set of the bosonic degrees of freedom we will encounter several massless fields in the form of differential p-forms (the so-called Ramond-Ramond forms). We will understand them as generalized electro-magnetic potentials that couple to D-branes just like the photon couples to electric particles. This will complete our pictures of D-branes which will finally be understood as extended dynamical objects with a mass (sourcing the gravitational field) and a charge (sourcing the Ramond-Ramond fields). In parallel we will also have a look at D-branes from the perspective of open superstring. Finally we will briefly mention the existence of the Type I theory and the two Heterotic theories. However we won't have time to do more than just mentioning them, to get the final picture of five distinct superstring theories.

In chapter 7 we will give an overview of string dualities. We will start with T-duality, which will be fully discussed at level of the worldsheet. We will then discuss the strong coupling behaviour of Type IIB and Type IIA which will bring us to (respectively) S-duality and M-Theory. We will finally see how the five superstring theories are all connected by perturbative and non-perturbative dualities, leaving us with a unique theory, whose fundamental structure and principles are however still rather mysterious.

## 1.4 Small guide through the literature

The material of these notes is taken for a large part from the classical String Theory books, to which the student is directed for a complete list of references. Students are highly encouraged to go through these books and not to be satisfied by these notes alone. An important source of material is the book by Blumenhagen, Lüst and Theisen [1], which is a modern and complete resource on the subject. The books by Polchinski [2, 3] are also highly recommended for the very insightful explanations, although they are updated up to 1997 or so, just before the $AdS/CFT$ correspondence. Anyway they are perfectly appropriate (although a bit advanced at first reading) for these lectures, which cover a subset of the two books. We have also used the classical books by Green, Schwarz and Witten [4,5]. These books however have been published in the 80's where some crucial elements (most notably D-branes) were still to be 'discovered'. As far as two dimensional conformal field theory is concerned, the classical reference for a self-contained introduction is the 'yellow book' by Di Francesco, Mathieu and Sénéchal [7]. The classical article by Friedan, Martinec and Shenker [6] is effectively a standard and pedagogical 'textbook' for the RNS formulation of the superstring. The lecture notes by David Tong [8] have also been a useful ingredient in the preparation of this course and students are encouraged to read them. For students who would like to be introduced to the world of string theory in a remarkable pedagogical way, which does not assume preexisting knowledge of QFT and GR we recommend the book by Zwiebach [9]. Other classical and useful resources on string theory are [10] and [11].

# 2 Free relativistic particle

The construction of a dynamical theory for a relativistic string is rather different from what one is used to in particle theory. For this reason, before attacking the string, we will discuss something which should be well known: a free relativistic particle. But we will do this using the same tools that we will later apply to the string.

## 2.1 Old covariant quantization

### 2.1.1 Classical aspects

A relativistic particle can be described by a one-dimensional field theory where the space-time coordinates are interpreted as $D$ Lorentz scalars $X^\mu$, labeled by the Lorentz indices $\mu = 0, 1, \ldots, D-1$, and depending only on the time parameter $\tau$. The dynamics of these fields is encoded in the world-line action[2]

$$S' = -m \int d\tau \sqrt{-\dot{X}^2} = -m \int d\tau \sqrt{-\eta_{\mu\nu} \frac{dX^\mu}{d\tau} \frac{dX^\nu}{d\tau}}, \tag{2.1}$$

which is proportional to the arc-length of the space-time trajectory. This action is clearly invariant under time reparametrization $\tau \to \tilde{\tau} = \tau + \theta(\tau)$, as it is easy to prove by insisting that $X$ transforms as a *scalar* in one dimension:

$$\tilde{X}^\mu(\tilde{\tau}) = X^\mu(\tau) \quad \Rightarrow \quad \frac{dX^\mu}{d\tau} = \frac{d\tilde{\tau}}{d\tau} \frac{d\tilde{X}^\mu}{d\tilde{\tau}}, \tag{2.2}$$

$$S[\tilde{X}^\mu(\tilde{\tau})] = -m \int d\tilde{\tau} \sqrt{-\dot{\tilde{X}}^2} = -m \int d\tau \frac{d\tilde{\tau}}{d\tau} \sqrt{-\left(\frac{d\tau}{d\tilde{\tau}}\right)^2 \dot{X}^2} = S[X^\mu(\tau)]. \tag{2.3}$$

This one-parameter diffeomorphism invariance is a gauge symmetry of the action. Related to this, the canonically conjugated momenta

$$P_\mu = \frac{\partial \mathcal{L}}{\partial \dot{X}^\mu} = \frac{m\dot{X}^\mu}{\sqrt{-\dot{X}^2}}, \tag{2.4}$$

are identically constrained to obey

$$P_\mu P^\mu + m^2 = 0. \tag{2.5}$$

Notice that, from a space-time perspective, this constraint is nothing but the mass shell condition for a free particle of mass $m$. An additional fact is that the associated Hamiltonian is identically vanishing:

$$H = P_\mu \dot{X}^\mu - \mathcal{L} = 0. \tag{2.6}$$

The fact that the canonical conjugated momenta are constrained, together with the vanishing of the Hamiltonian makes the quantization procedure (although not impossible) rather intricate. Moreover if we look at the action (2.1) as a one-dimensional field theory, we don't recognize (because of the square root) a standard kinetic term for the scalar fields $X^\mu(\tau)$.

To solve all of these problems at once we introduce a new classically equivalent action which contains a new one-dimensional field $e(\tau)$ in addition to the scalar fields $X^\mu$:

$$S[X, e] = \frac{1}{2} \int d\tau \left( \frac{\dot{X}^2}{e} - em^2 \right). \tag{2.7}$$

---

[2]We always use the metric signature $(-, +, \ldots, +)$.

The new field $e(\tau)$ is the "ein-bein".[3] In this trivial 1-dimensional case we can write $e = \sqrt{-g_{\tau\tau}}$ and rewrite the action $S[X^\mu, e]$ as

$$S[X^\mu, g_{\tau\tau}] = -\frac{1}{2} \int d\tau \, \sqrt{-g}\big(g^{\tau\tau}\dot{X}^\mu \dot{X}_\mu + m^2\big), \tag{2.8}$$

namely the action of $D$ scalar fields on a one dimensional manifold whose metric $g_{\tau\tau}$ is considered as a dynamical field. We have also used the trivial 1-dimensional identification

$$g^{\tau\tau} = \frac{1}{g_{\tau\tau}}, \quad g = \det g = g_{\tau\tau}. \tag{2.9}$$

This action describes $D$ free scalar fields $X^\mu$ covariantly coupled to a one-dimensional metric $g_{\tau\tau}$ and is therefore obviously invariant under $\tau$-reparametrizations (which are just the one-dimensional diffeomorphisms). Using the general formula for the change of a field $\phi$ under an infinitesimal reparametrization $\tilde{\tau} = \tau + \theta(\tau)$

$$\tilde{\phi}(\tilde{\tau}) = \phi(\tau) + \delta\phi(\tau) + \dot{\phi}(\tau)\delta\tau \equiv \tilde{\phi}(\tau) + \dot{\phi}(\tau)\theta(\tau), \tag{2.10}$$

and the fact that the einbein transforms as a 1-form

$$\tilde{e}(\tilde{\tau})d\tilde{\tau} = e(\tau)d\tau, \tag{2.11}$$

we can write the complete set of gauge transformations (1-dimensional diffeos) as

$$\delta e = -\frac{d(e\theta)}{d\tau}, \tag{2.12a}$$

$$\delta X^\mu = -\theta\dot{X}(\tau), \tag{2.12b}$$

$$\delta\tau = \theta(\tau). \tag{2.12c}$$

It is easy to verify that (2.7) is invariant under this set of transformations.

> **Exercise 2.1.1**
>
> Check eqs. (2.12a), (2.12b) and (2.12c).

> **Exercise 2.1.2**
>
> Verify that (2.7) is invariant under the transformations (2.12a), (2.12b) and (2.12c).

If we now evaluate the equations of motion (EOM) for $e$ and $X^\mu$ we obtain

$$\frac{\delta S}{\delta e} = 0 \implies \dot{X}^2 + e^2 m^2 = 0, \tag{2.13a}$$

$$\frac{\delta S}{\delta X^\mu} = 0 \implies \frac{d}{d\tau}\frac{\dot{X}^\mu}{e} = 0, \tag{2.13b}$$

so that replacing (2.13a) in (2.7) we obtain the starting action $S'$ (2.1), which proves the equivalence of the two actions. Moreover from equation (2.13a) we recover exactly the constraint (2.5) over the momentum $P_\mu = \dot{X}_\mu/e$. *The constraint (2.5), giving rise to the physical mass-shell condition, is now realized as the equation of motion for the ein-bein $e$.*

---

[3]A $D$-dimensional (pseudo-) Riemannian manifold which is locally Minkowski has a metric $g_{\mu\nu}$ which can be written $g_{\mu\nu} = e^a_\mu \eta_{ab} e^b_\nu$, where $e^a_\mu$ are the components of $D$ one-forms $e^a = e^a_\mu dx^\mu$ (with $a = 0, \cdots, D-1$)) which are usually called *vielbeins*.

### 2.1.2 Quantization

We may now proceed to the first quantization method, equivalent to the Gupta-Bleuer method in QED, that we will refer to as *old covariant quantization* (OCQ). The first step to implement this method is to consider $e$ as a gauge parameter which has to be fixed. We will thus set $e$ to a *fiducial* value:

$$e = \hat{e} = 1 \quad \text{(fiducial value)}. \tag{2.14}$$

We may convince ourselves that this fiducial value can be achieved, starting from *any* $e(\tau)$, by a reparametrization (2.12a):

$$\delta e = -\partial_\tau(\theta e) = 1 - e. \tag{2.15}$$

Explicitly:

$$\theta(\tau) = -\frac{1}{e}\left((\tau - \tau_0) - \int_{\tau_0}^{\tau} d\tau' e(\tau')\right). \tag{2.16}$$

> **Exercise 2.1.3**
>
> Check eq. (2.16).

The gauge fixed action is given by simply setting $e = 1$ in (2.7) and reads

$$S^{\text{gauge-fixed}}(X) = S(X, e)\Big|_{e=1} = \frac{1}{2}\int d\tau \left(\dot{X}^2 - m^2\right). \tag{2.17}$$

If we vary this action with respect to $X$ we get

$$\ddot{X}^\mu = 0. \tag{2.18}$$

This equation describes a generic geodesic motion in flat space-time but there is no information at all about the mass-shell of the particle! In other words, if we consider the canonical momentum

$$P_\mu = \dot{X}_\mu, \tag{2.19}$$

it seems we have lost the mass-shell $P^2 + m^2 = 0$. What has happened? The 'sin' we have committed has been to fix $e = 1$ and then to forget the equation of motion for $e$. The missing equation is given by (2.13a) with $e \to 1$:

$$\dot{X}^2 + m^2 = 0 \quad \longleftrightarrow \quad P^2 + m^2 = 0. \tag{2.20}$$

Notice that this cannot be obtained anymore by varying the gauge-fixed action (2.17) but we have to carry it as an additional constraint on the space of solutions to (2.18).

Now, armed with this understanding, we can quantize the theory. We follow *canonical quantization* which means that we impose equal-time canonical commutation relations on the space of solutions of (2.18). The generic solution to (2.18) is given by

$$X^\mu(\tau) = x^\mu + p^\mu \tau, \tag{2.21a}$$
$$P^\mu(\tau) = \dot{X}^\mu = p^\mu. \tag{2.21b}$$

In the quantum world the *solutions $X$ and $P$* (not the generic configurations of $X$ and $P$ not solving the X-eom!) become canonically conjugated operators with respect to the *equal time* commutator[4]

$$[\hat{X}^\mu(\tau), \hat{P}^\nu(\tau)] = i(\hbar)\eta^{\mu\nu}, \tag{2.22}$$

---

[4]From now on we will set $\hbar = c = 1$.

which explicitly means

$$[\hat{x}^\mu, \hat{p}^\nu] = i\eta^{\mu\nu}. \tag{2.23}$$

This is an Heisenberg algebra which is realized in a Hilbert space of momentum eigenstates:

$$\hat{p}^\mu|0,p\rangle = p^\mu|0,p\rangle, \tag{2.24}$$

$$\langle 0,p|0,p'\rangle = \delta^d(p-p'). \tag{2.25}$$

The generic state in this space is

$$|\Psi\rangle = \int d^d p\, \psi(p)|0,p\rangle. \tag{2.26}$$

The coefficient $\psi(p)$ (or equivalently its Fourier transform $\tilde{\psi}(x) = \int dp\, \psi(p)e^{ip\cdot x}$) is a scalar field in space-time, not subject to any equation of motion. Therefore the obtained Hilbert space contains the most generic *off-shell* configurations for a scalar field in space time. The requirement of physicality (mass-shell condition) comes from the missing equation of motion of the ein-bein $e$. Classically the equation is simply $p^2 + m^2 = 0$. In the quantum world we have to impose that the operator $\hat{p}^2 + m^2$ vanishes when inserted between physical states $|0,p\rangle^{(phys)}$

$$^{(phys)}\langle 0,p'|(\hat{p}^2 + m^2)|0,p\rangle^{(phys)} = 0 \qquad \text{(Physical state condition)}. \tag{2.27}$$

Thus a state will be *physical* if and only if

$$\boxed{\left(\hat{p}^2 + m^2\right)|\text{phys}\rangle = 0.} \tag{2.28}$$

If we now use the Fourier transform to change basis to the position eigenvectors, we get

$$\psi(p) = \int d^d x\, \tilde{\psi}(x)e^{-ip\cdot x}. \tag{2.29}$$

Then the physical condition is translated to

$$\boxed{(\Box - m^2)\tilde{\psi}(x) = 0,} \tag{2.30}$$

so we find that the wave function must satisfy the Klein-Gordon equation.

## 2.2  BRST quantization

The old covariant quantization we have gone through in the last section is perfectly fine for the particle but, as we will see, it falls somehow short for the string in determining unambiguosly the critical dimension and the correct physical spectrum. To see these two crucial things neatly emerge, we need to consider the BRST quantization (or the related light-cone quantization). In this preliminary section we are going to review the BRST formalism. This is the 'correct' method to quantize theories having gauge symmetries controlled by a Lie algebra structure. As we will see this is sufficient for both the particle and the string.

To describe the BRST quantization we consider a general action $S[\phi_i]$ for a set of fields $\phi_i$.[5] This action is equipped with a gauge symmetry (redundancy) given by the following fields transformations:

$$\delta \phi_i = \xi^a \delta_a \phi_i \,, \tag{2.33}$$

where $\xi^a$ are the gauge parameters associated to the algebra generators $\delta_a$ which we will assume to satisfy the properties of a Lie algebra:

$$[\delta_a, \delta_b] = f_{ab}{}^c \delta_c \,. \tag{2.34}$$

We define the (Minkowskian) path integral

$$Z = \int \frac{\mathcal{D}\phi_i}{\text{Vol}(\mathcal{G})} e^{iS[\phi_i]} \,, \tag{2.35}$$

which is naturally normalized by the volume of the gauge group. It is clear that this last quotient is necessary to avoid the field overcounting due to the gauge redundancy. Indeed, if we integrate over all the possible field configurations we will also consider the states that just differ by a gauge transformation and therefore have identical action contribution. Given the fact that the gauge equivalent states are infinite this produces a divergence in the path integral, proportional to the volume of the gauge group. However this divergence is not physical because gauge equivalent states are physically equivalent and we must implement a gauge fixing procedure to explicitly factorize and simplify away the group volume. The gauge fixing can be implemented in the case of a Lie Group using the *Faddeev-Popov method* (FP).

Let us consider a gauge fixing condition $F^A(\phi_i) = 0$ [6] that defines the gauge nonequivalent states $\hat{\phi}_i \mid F^A(\hat{\phi}_i) = 0$. By definition it is possible then to describe every other state in the field space using the $\hat{\phi}_i$ gauge orbits, and in particular we can reparametrize the fields $\phi_i$ as $\phi_i = \hat{\phi}_i^\xi$ with the condition that under infinitesimal gauge transformation the parametrization behaves as $\hat{\phi}_i^\xi \sim \hat{\phi}_i + \xi^a \delta_a \hat{\phi}_i$.

The first step to implement FP is to manipulate the trivial functional identity described as a functional integral over the (functional) gauge group, namely

$$\int \mathcal{D}\xi^a \delta(\xi^a) = 1 \,, \tag{2.36}$$

using the well known property of the Dirac delta $\delta(f(x))|f'(x_0)| = \delta(x - x_0)$ (where $x_0$ is a simple zero of $f(x)$), and generalizing it to the functional integral

$$\int \mathcal{D}\xi^a \delta(\xi^a) = \int \mathcal{D}\xi^a \det\left[ \frac{\delta F^A(\phi_i^\xi)}{\delta \xi^a} \right]_{\xi=0} \delta(F^A(\phi_i^\xi)) = 1 \,. \tag{2.37}$$

---

[5]In our notation the indices will be used to denote every parameter of the field $\phi$, both continuous (like the space-time coordinates) and discrete. The saturation of two indices must be interpreted as an integration in the continuous case and as a summation in the discrete case:

$$\phi_i \varphi^i = \sum_i \phi_i \varphi_i \,, \tag{2.31}$$

$$\phi_a \varphi^a = \int da\, \phi(a)\varphi(a) \,. \tag{2.32}$$

As an example, in the free particle case $\phi_i = (X^\mu(\tau), e(\tau))$ so that the $i$ index represents $\tau, \mu$ and also distinguishes between $X$ and $e$.

[6]Lowercase and uppercase Latin letters will be used to identify respectively the gauge group and the gauge fixing functions indices.

Using this resolution of the identity we can then rewrite 2.35 as

$$Z = \int \frac{\mathcal{D}\phi_i}{\text{Vol}(\mathcal{G})} e^{iS[\phi_i]} \int \mathcal{D}\xi^a \det\left[\frac{\delta F^A(\phi_i^\xi)}{\delta \xi^a}\right]_{\xi=0} \delta(F^A(\phi_i^\xi)). \tag{2.38}$$

We can use the gauge invariance of the action and of the measure to transform the field as $\phi_i \to \phi_i^\xi$ in the action and in the measure so that we can then globally rename $\phi_i^\xi \to \phi_i$ so that the full integrand is $\xi$ independent and we can then factorize the integral over $\xi^a$ as

$$Z = \int \mathcal{D}\xi^a \int \frac{\mathcal{D}\phi_i}{\text{Vol}(\mathcal{G})} e^{iS[\phi_i]} \det\left[\frac{\delta F^A(\phi_i^\xi)}{\delta \xi^a}\right]_{\xi=0} \delta(F^A(\phi_i)). \tag{2.39}$$

By definition $\text{Vol}(\mathcal{G}) = \int \mathcal{D}\xi^a$ and the factorized integral simplifies away with the denominator, leaving a normalized path integral with the integrand explicitly defined to constrain the fields over the surface $F(\phi_i) = 0$ which imposes the gauge fixing condition. The new two factors in the path integral can be further manipulated and included in the exponential using the following two properties:

- The Dirac delta can be rewritten in an exponential form by introducing a new bosonic auxiliary field $B_A$:

$$\delta(F^A(\phi_i)) = \int \mathcal{D}B_A \, e^{iB_A F^A(\phi)}. \tag{2.40}$$

- To find an exponential form for the determinant we can recall the properties of the Berezin integral for Grassmann numbers. Given a square matrix $M_{\alpha\beta}$ we have that

$$\int db_1 dc_1 \ldots db_m dc_m \, e^{-b^\alpha M_{\alpha\beta} c^\beta} = \det M. \tag{2.41}$$

We recall that a Grassmann number $\theta$ is an anticommuting object obeying $\theta^2 = 0$. The Berezin integral is thus defined to be

$$\int d\theta \, \theta^n = \delta_{1,n}, \tag{2.42}$$

$$\int d\theta_1 d\theta_2 \, \theta_2 \theta_1 = 1. \tag{2.43}$$

In the end we obtain the gauge fixed path integral

$$Z = \int \mathcal{D}\phi_i \mathcal{D}B_A \mathcal{D}b_A \mathcal{D}c^a \, e^{i\left[S[\phi_i] + B_A F^A + i b_A \frac{\delta F^A}{\delta \xi^a} c^a\right]}, \tag{2.44}$$

where $\phi_i$ are the original physical fields, $B_A$ is an auxiliary bosonic field acting as a Lagrange multiplier for the gauge fixing condition $F^A(\phi) = 0$ and $c^a, b_A$ are new fermionic fields known as Fadeev-Popov ghosts divided respectively in ghosts $c^a$ and anti-ghosts $b_A$. The resulting exponent

$$S_{\text{FP}}[\phi_i, B_A, c_a, b_A] = S[\phi_i] + B_A F^A + i b_A \frac{\delta F^A}{\delta \xi^a} c^a, \tag{2.45}$$

is the gauge fixed action of the theory. Assuming that the gauge fixing condition $F^A(\phi) = 0$ fixes the gauge completely, the path integral of this action will now integrate over physically inequivalent configurations and it is therefore the one that we have to use to compute physical quantities, for example correlation functions of gauge invariant operators.

Since we have fixed the gauge we are not supposed to have anymore a gauge symmetry in the FP action (2.45). But in fact we have now a new *global* and *fermionic* symmetry which is a remnant of the local gauge symmetry:

$$\delta_B \phi_i = c^a \delta_a \phi \,, \tag{2.46a}$$

$$\delta_B B_A = 0 \,, \tag{2.46b}$$

$$\delta_B b_A = i B_A \,, \tag{2.46c}$$

$$\delta_B c^a = \frac{1}{2} f^a_{\ bc} c^b c^c \,. \tag{2.46d}$$

$\delta_B$ generates the Becchi-Rouet-Stora-Tyutin (BRST) symmetry. We will explicitly check the invariance of the action in a while, using an elegant quick method. But you may want to check it directly by hand. Notice that $\delta_B$ relates a boson $\phi$ to the a fermion $c$ and it is therefore a *fermionic* symmetry (in this sense it is quite similar to *supersymmetry* with the important difference that the parameter is a fermion but not a spinor). Notice also that this set of transformations don't depend on the choice of gauge-fixing $F_A(\phi)$ (whereas the FP action (2.45) obviously does). Moreover this symmetry is *nilpotent*, namely it satisfies

$$\delta_B^2 = 0 \,. \tag{2.47}$$

To prove this we just need to prove that the action of $\delta_B^2$ is zero on every field, in fact if we take a generic functional of the fields $F[\varphi_I]$, where $\varphi_I$ represents a field in the set $\{\phi_i, B_A, b_A, c^a\}$ and $|\varphi_i|$ its Grassmannality, we obtain

$$\delta_B^2 F[\varphi_I] = \delta_B^2 \varphi_I \frac{\delta F}{\delta \varphi_I} + (-1)^{|\varphi_I|+1} \delta_B \varphi_I \delta_B \varphi_J \frac{\delta^2 F}{\delta \varphi_J \delta \varphi_I}$$

$$= \frac{1}{2} \left[ (-1)^{|\varphi_I|+1} \delta_B \varphi_I \delta_B \varphi_J \frac{\delta^2 F}{\delta \varphi_J \delta \varphi_I} + (-1)^{|\varphi_J|+1} \delta_B \varphi_J \delta_B \varphi_I \frac{\delta^2 F}{\delta \varphi_I \delta \varphi_J} \right] + \delta_B^2 \varphi_I \frac{\delta F}{\delta \varphi_I}$$

$$= \frac{1}{2} \left[ (-1)^{|\varphi_I|+1} \delta_B \varphi_I \delta_B \varphi_J \frac{\delta^2 F}{\delta \varphi_J \delta \varphi_I} + (-1)^{|\varphi_I|} \delta_B \varphi_I \delta_B \varphi_J \frac{\delta^2 F}{\delta \varphi_J \delta \varphi_I} \right] + \delta_B^2 \varphi_I \frac{\delta F}{\delta \varphi_I}$$

$$\implies \delta_B^2 F[\varphi_I] = \delta_B^2 \varphi_I \frac{\delta F}{\delta \varphi_I} \,, \tag{2.48}$$

and in the end $\delta_B^2 F[\varphi_I] = 0 \ \forall F \iff \delta_B^2 \varphi_I = 0$. The result of $\delta_B^2$ on the fields can be proven to be zero by direct evaluation:

$$\delta_B^2 B_A = \delta_B 0 = 0 \,, \tag{2.49}$$

$$\delta_B^2 b_A = i \delta_B B_A = 0 \,, \tag{2.50}$$

$$\delta_B^2 \phi_i = \delta_B(c^a \delta_a \phi_i) = \delta_B c^a \delta_a \phi_i - c^a \delta_B(\delta_a \phi_i) = \frac{1}{2} f^a_{\ bc} c^b c^c \delta_a \phi_i - c^a c^b \delta_a \delta_b \phi_i$$

$$= \frac{1}{2} f^a_{\ bc} c^b c^c \delta_a \phi_i - \frac{1}{2} \left[ c^a c^b \delta_a \delta_b \phi_i + c^b c^a \delta_b \delta_a \phi_i \right]$$

$$= \frac{1}{2} f^a_{\ bc} c^b c^c \delta_a \phi_i - \frac{1}{2} c^a c^b [\delta_a, \delta_b] \phi_i = \frac{1}{2} f^a_{\ bc} c^b c^c \delta_a \phi_i - \frac{1}{2} f^c_{\ ab} c^a c^b \delta_c \phi_i$$

$$= 0 \,, \tag{2.51}$$

$$\delta_B^2 c^a = \frac{1}{2} f^a_{\ bc} \delta_B(c^b c^c) = \frac{1}{2} f^a_{\ bc} \delta_B(c^b) c^c - \frac{1}{2} f^a_{\ bc} c^b \delta_B(c^c)$$

$$= \frac{1}{4} f^a_{\ bc} f^b_{\ de} c^d c^e c^c - \frac{1}{4} f^a_{\ bc} f^c_{\ de} c^b c^d c^e = \frac{1}{4} f^a_{\ bc} f^b_{\ de} c^d c^e c^c - \frac{1}{4} f^a_{\ cb} f^b_{\ de} c^c c^d c^e$$

$$= \frac{1}{2} f^a_{\ bc} f^b_{\ de} c^c c^d c^e = \frac{1}{6} f^a_{\ b[c} f^b_{\ de]} c^c c^d c^e$$

$$= 0 \,, \tag{2.52}$$

where the last term is zero due to the Jacobi identity. We can now prove that $\delta_B$ is an actual symmetry of $S_{FP}$. To do this we first observe that

$$\delta_B(-ib_A F^A) = B_A F^A + ib_A \delta_B F^A = B_A F^A + ib_A \delta_B \phi_i \frac{\delta F^A}{\delta \phi_i}$$

$$= B_A F^A + ib_A \delta_a \phi_i \frac{\delta F^A}{\delta \phi_i} c^a = B_A F^A + ib_A \frac{\delta F^A}{\delta \xi^a} c^a. \qquad (2.53)$$

This means that the total FP action differs from the initial gauge-invariant action by a $\delta_B$-variation. This is sufficient to establish the BRST invariance of the FP lagrangian

$$\delta_B S_{FP} = \delta_B \left[ S[\phi_i] + \delta_B(-ib_A F^A) \right] = \delta_B S[\phi_i] + \delta_B^2(-ib_A F^A) = c^a \delta_a S[\phi_i] = 0, \qquad (2.54)$$

which is zero because the original action is invariant under gauge transformation. Thanks to the Noether theorem we know then that there exists a conserved fermionic charge $Q_B$, the *BRST charge*.

In the quantum theory $Q_B$ will be an operator which will realize the BRST transformations on other fields/operators via a properly defined adjoint action

$$\delta_B \varphi_i = i[\varphi_i, Q_B] = i(-1)^{|\varphi|+1}[Q_B, \varphi_i], \qquad (2.55)$$

where $[A, B] = AB - (-1)^{|A||B|} BA$ is the graded commutator and $|A|$ is the grassmanality of $A$. Since $\delta_B^2 = 0$, $Q_B$ will also be nilpotent[7]:

$$0 = \delta_B^2 \varphi_i = [Q_B, [Q_B, \varphi_i]]$$
$$= Q_B(Q_B \varphi_i - (-1)^{|\varphi_i|} \varphi_i Q_B) - (-1)^{|\varphi_i|+1}(Q_B \varphi_i - (-1)^{|\varphi_i|} \varphi_i Q_B) Q_B$$
$$= Q_B^2 \varphi_i - \varphi_i Q_B^2 = [Q_B^2, \varphi_i] \quad \forall \varphi_i$$

$$\implies Q_B^2 = 0. \qquad (2.56)$$

Since $Q_B$ is a nilpotent linear operator, it defines a cohomological structure on the fields Hilbert space $\mathcal{H}$. In particular we can define *exact states* as $|E\rangle = Q_B |\Psi\rangle$, which are automatically inside the kernel of $Q_B$, and *closed states* which are in the kernel of $Q_B$, $Q_B |C\rangle = 0$ without necessarily being exact. Given the definition of exact states we can then define an equivalence relation $\sim$ in $\mathrm{Ker} Q_B$ as

$$|\Psi\rangle \sim |\Psi\rangle + Q_B |\Phi\rangle, \quad |\Psi\rangle \in \mathrm{Ker} Q_B, \qquad (2.57)$$

which means that two states are equivalent iff they differ by an exact state. The cohomology of the operator $Q_B$ is then defined as the quotient

$$\mathrm{H}(Q_B) = \mathrm{Ker} Q_B / \mathrm{Im} Q_B. \qquad (2.58)$$

The existence of the cohomology is the key point of the formalism. Indeed physical states in the Hilbert space should be identified with the cohomology of $Q_B$, as we are now going to argue. We start by the following consideration: an arbitrary amplitude between two physical states $|i\rangle = \varphi_i |0\rangle$, $|f\rangle = \varphi_f |0\rangle$ can be evaluated using the path integral

$$\langle f | i \rangle = \int \mathcal{D}\varphi \, \varphi_f \varphi_i \, e^{iS_{GF}}, \qquad (2.59)$$

---

[7]Importantly assuming that there are no *anomalies* in the process of quantization!

and it must be invariant under a change of gauge fixing $F_A \to F_A + \delta F_A$. This means that

$$
\begin{aligned}
\delta_{GF} \langle f | i \rangle &= \int \mathcal{D}\varphi \, \varphi_f \varphi_i \, e^{i\left(S_{GF} + \delta_B(-ib_A\delta F^A)\right)} - \int \mathcal{D}\varphi \, \varphi_f \varphi_i \, e^{iS_{GF}} \\
&= \int \mathcal{D}\varphi \, \varphi_f \, \delta_B(b_A\delta F^A) \varphi_i \, e^{-S_{GF}} = \langle f | \delta_B(b_A\delta F^A) | i \rangle \\
&= i \langle f | [b_A\delta F^A, Q_B] | i \rangle = 0 \,,
\end{aligned}
\tag{2.60}
$$

and since it must be zero $\forall \, \delta F_A$ this imposes that $Q_B | i \rangle = \langle f | Q_B = 0$. This means that a state is physical only if it is closed under the action of $Q_B$:

$$
Q_B | \text{phys} \rangle = 0 \,.
\tag{2.61}
$$

On the other end if we take a closed state that is also exact $|E\rangle = Q_B |\Lambda\rangle$ and we evaluate its amplitude with a physical state $|\phi\rangle$, we obtain

$$
\langle E | \phi \rangle = \langle \Lambda | Q_B | \phi \rangle = 0 \,, \quad \text{because} \quad Q_B | \phi \rangle = 0 \,,
\tag{2.62}
$$

which means that it is impossible to transit between an exact state and a physical state. This allow us to conclude that the physical states are identified with the cohomology of $Q_B$.

## 2.3 BRST quantization of the free particle

We can now apply the BRST formalism to the free particle case considering the classical action

$$
S[X^\mu, e] = \frac{1}{2} \int d\tau \left( \frac{\dot{X}^2}{e} - m^2 e \right) .
\tag{2.63}
$$

The gauge symmetry is represented by the reparametrization invariance $\tau \to \tau + \theta(\tau)$, namely the group of one dimensional diffeomorphisms. The infinitesimal gauge transformations are

$$
\delta X^\mu = -\theta(\tau) \partial_\tau X^\mu \,,
\tag{2.64}
$$
$$
\delta e = -\partial_\tau (e\theta(\tau)) \,.
\tag{2.65}
$$

The compact notation we used in the previous section should then be understood as

$$
\phi_i \to \{ X^\mu(\tau), e(\tau) \} \,,
\tag{2.66}
$$
$$
\xi^a \to \theta(\tau') \,.
\tag{2.67}
$$

With this understanding we have[8]

$$
\begin{aligned}
\xi^a \delta_a X^\mu(\tau) &\to \int d\tau' \, \theta(\tau') \delta_{\tau'} X^\mu(\tau) = -\theta(\tau) \partial_\tau X^\mu(\tau) \\
&\implies \delta_{\tau'} X^\mu(\tau) = -\delta(\tau - \tau') \partial_\tau X^\mu(\tau) \,,
\end{aligned}
\tag{2.68}
$$

---

[8]The derivative of the Dirac $\delta$ is functionally defined by its integral action on a function. In particular:

$$
\int dx \, \partial_x \delta(x - x_0) f(x) = \lim_{h \to 0} \int dx \, \frac{\delta(x + h - x_0) - \delta(x - x_0)}{h} f(x) = \lim_{h \to 0} \frac{f(x_0 - h) - f(x_0)}{h} = -f'(x_0) \,,
$$
$$
\int dx \, \partial_{x_0} \delta(x - x_0) f(x) = \lim_{h \to 0} \int dx \, \frac{\delta(x - h - x_0) - \delta(x - x_0)}{h} f(x) = \lim_{h \to 0} \frac{f(x_0 + h) - f(x_0)}{h} = f'(x_0) \,.
$$

In the text we denote $\delta'(x) \equiv \partial_x \delta(x)$ which obeys $\delta'(-x) = -\delta'(x)$.

$$\xi^a \delta_a e(\tau) \to \int d\tau' \, \theta(\tau') \delta_{\tau'} e(\tau) = -\frac{d(\theta e)}{d\tau}(\tau)$$

$$\implies \delta_{\tau'} e(\tau) = \delta'(\tau' - \tau) e(\tau'), \tag{2.69}$$

so that we have defined the infinitesimal generators of the group. Now we need to find the values of the structure constants $f^a{}_{bc}$, that can be derived evaluating the action of the commutator $[\delta_{\tau_1}, \delta_{\tau_2}]$ on $X^\mu(\tau)$:

$$[\delta_{\tau_1}, \delta_{\tau_2}] X^\mu(\tau) = [\delta(\tau - \tau_1)\partial_\tau \delta(\tau - \tau_2) - \delta(\tau - \tau_2)\partial_\tau \delta(\tau - \tau_1)]\partial_\tau X^\mu(\tau) \tag{2.70}$$

$$= \int d\tau_3 \, f^{\tau_3}{}_{\tau_1 \tau_2} \delta_{\tau_3} X^\mu(\tau),$$

$$\implies f^{\tau_3}{}_{\tau_1 \tau_2} = \delta(\tau_3 - \tau_1)\delta'(\tau_3 - \tau_2) - \delta(\tau_3 - \tau_2)\delta'(\tau_3 - \tau_1). \tag{2.71}$$

---

**Exercise 2.3.1**

Obtain the same structure constants by computing $[\delta_{\tau_1}, \delta_{\tau_2}] e(\tau)$ and, in doing this, realize that we can equivalently write

$$f^{\tau_3}{}_{\tau_1 \tau_2} = \delta'(\tau_1 - \tau_2)(\delta(\tau_1 - \tau_3) + \delta(\tau_2 - \tau_3)). \tag{2.72}$$

---

Now we can write down the BRST transformations from the general relations (2.46a), (2.46b), (2.46c) and (2.46d):

$$\delta_B X^\mu = [c^a \delta_a X^\mu] = \int d\tau_1 \, c(\tau_1) \delta_{\tau_1} X^\mu(\tau) = -c\dot{X}^\mu, \tag{2.73a}$$

$$\delta_B e = [c^a \delta_a e] = \int d\tau_1 \, c(\tau_1) \delta_{\tau_1} e(\tau) = -\frac{d(ce)}{d\tau}, \tag{2.73b}$$

$$\delta_B b = iB, \tag{2.73c}$$

$$\delta_B c = \left[\frac{1}{2} f^a{}_{bc} c^b c^c\right] = \frac{1}{2} \int d\tau_1 d\tau_2 \, f^\tau{}_{\tau_1 \tau_2} c(\tau_1) c(\tau_2)$$

$$= \frac{1}{2} \int d\tau_1 d\tau_2 \left[\delta(\tau - \tau_1)\delta'(\tau - \tau_2) - \delta(\tau - \tau_2)\delta'(\tau - \tau_1)\right] c(\tau_1) c(\tau_2)$$

$$= \frac{1}{2}[-c\dot{c} + \dot{c}c] = -c\dot{c}, \tag{2.73d}$$

which is nilpotent.

---

**Exercise 2.3.2**

Check the explicit form of these transformations and check that they are indeed nilpotent. In doing this pay attention to the fact that $\delta'(x - y) = \frac{1}{2}(\partial_x - \partial_y)\delta(x - y)$.

---

Given the gauge fixing condition $F(\tau) = e(\tau) - 1 = 0 \to e = 1$ we can then evaluate the gauge fixed action from the general result (2.45):

$$
\begin{aligned}
S_{GF} &= S[X^\mu, e] + B_A F^A + i b_A \delta_a F^A c^a \\
&= S[X^\mu, e] + \int d\tau\, B(\tau)(e(\tau) - 1) + i \int d\tau\, b(\tau) c^a \delta_a (e(\tau) - 1) \\
&= \int d\tau \left[ \frac{1}{2} \left( \frac{\dot{X}^2}{e} - em^2 \right) + B(e-1) - ib \frac{d(ce)}{d\tau} \right] \\
&= \int d\tau \left[ \frac{1}{2} \left( \frac{\dot{X}^2}{e} - em^2 \right) + B(e-1) + i\dot{b}ce \right].
\end{aligned}
\tag{2.74}
$$

The gauge fixed path integral is then immediately obtained using $S_{GF}$:

$$
\begin{aligned}
Z &= \int \mathcal{D}X^\mu \mathcal{D}e \mathcal{D}B \mathcal{D}b \mathcal{D}c\, e^{i \int d\tau \left[ \frac{1}{2} \left( \frac{\dot{X}^2}{e} - em^2 \right) + B(e-1) + i\dot{b}ce \right]} \\
&= \int \mathcal{D}X^\mu \mathcal{D}e \mathcal{D}b \mathcal{D}c\, e^{i \int d\tau \left[ \frac{1}{2} \left( \frac{\dot{X}^2}{e} - em^2 \right) + i\dot{b}ce \right]} \delta(e-1) \\
&= \int \mathcal{D}X^\mu \mathcal{D}b \mathcal{D}c\, e^{i \int d\tau \left[ \frac{1}{2} (\dot{X}^2 - m^2) + i\dot{b}c \right]},
\end{aligned}
\tag{2.75}
$$

where we have integrated out the fields $e$ and $B$, obtaining the reduced BRST action

$$
S_R[X, b, c] = \int d\tau \left[ \frac{1}{2} (\dot{X}^2 - m^2) + i\dot{b}c \right],
\tag{2.76}
$$

where $e$ is fixed to 1 while $B$ is determined using the equation of motion of $e$:

$$
\begin{aligned}
\frac{\delta S_{GF}}{\delta e} \Big|_{e=1} &= \left[ B + i\dot{b}c - \frac{1}{2} \left( \frac{\dot{X}^2}{e^2} + m^2 \right) \right]_{e=1} \\
&= B + i\dot{b}c - \frac{1}{2} (\dot{X}^2 + m^2) = 0 \\
\implies B &= \frac{1}{2} (\dot{X}^2 + m^2) - i\dot{b}c.
\end{aligned}
\tag{2.77}
$$

It is important to notice that the reduced action still contains a residual BRST symmetry given by

$$
\delta_B X^\mu = -c\dot{X}^\mu,
\tag{2.78a}
$$

$$
\delta_B c = -c\dot{c},
\tag{2.78b}
$$

$$
\delta_B b = -iB = i\left( \frac{1}{2} (\dot{X}^2 + m^2) - i\dot{b}c \right),
\tag{2.78c}
$$

which is also nilpotent. However since we have used the $e$ and $B$-eoms, this transformation leaves the action invariant and is nilpotent only when all the remaining eoms are imposed:

$$
\frac{\delta S}{\delta b} = \dot{c} = 0 \implies c(\tau) = c_0,
\tag{2.79a}
$$

$$
\frac{\delta S}{\delta c} = \dot{b} = 0 \implies b(\tau) = b_0,
\tag{2.79b}
$$

$$
\frac{\delta S}{\delta X^\mu} = \ddot{X}^\mu = 0 \implies X^\mu(\tau) = x^\mu + p^\mu \tau.
\tag{2.79c}
$$

> **Exercise 2.3.3**
>
> 1) Check that the reduced BRST transformations leaves the action (2.76) invariant, using the eoms.
> 2) Check that the reduced BRST transformations are nilpotent, using the eoms.

To have a nilpotent BRST symmetry only realized when the fields are on-shell is not a problem for us because, to quantize the theory, we work on the space of solutions. We thus proceed to canonical quantization of the reduced BRST action (2.76).

To do so we evaluate the conjugate momenta

$$P_\mu(\tau) = \frac{\partial \mathcal{L}}{\partial \dot{X}^\mu} = \dot{X}_\mu = p_\mu, \tag{2.80a}$$

$$P_c(\tau) = \frac{\partial \mathcal{L}}{\partial \dot{c}} = -ib = -ib_0, \tag{2.80b}$$

$$P_b(\tau) = \frac{\partial \mathcal{L}}{\partial \dot{b}} = -ic = -ic_0, \tag{2.80c}$$

that are realized on the Hilbert space $\mathcal{H}$ as the usual self-adjoint operators satisfying the canonical commutation relations

$$[X^\mu(\tau), P_\nu(\tau)] = [x^\mu, p_\nu] = i\delta^\mu_\nu, \tag{2.81a}$$

$$[c, b] = [c_0, b_0] = 1, \tag{2.81b}$$

with

$$x^\dagger = x, \qquad P^\dagger = P, \qquad c_0^\dagger = c_0, \qquad b_0^\dagger = b_0. \tag{2.82}$$

Notice that canonical commutation relations for the fermionic fields $(b, c)$ do not contain the $i$, because the anti-commutator of hermitian objects is already hermitian (the same happens when quantizing the Dirac lagrangian). Since ghost and matter fields obviously commute, the Hilbert space can be decomposed in the tensor product of two disjoint spaces $\mathcal{H} = \mathcal{H}_{\text{Matter}} \otimes \mathcal{H}_{\text{Ghosts}} = \mathcal{H}_M \otimes \mathcal{H}_G$ that can be studied separately.

- $\mathcal{H}_M = \text{Span}\{|0, p\rangle\}$, which is the matter Hilbert space of the old covariant quantization which we have already studied.

- $\mathcal{H}_G$ which is spanned by acting $b_0$ and $c_0$ operators on a vacuum $|\downarrow\rangle$, which is defined by $b_0 |\downarrow\rangle = 0$. We can define the state $|\uparrow\rangle \equiv c_0 |\downarrow\rangle$. Because of the nilpotency of $c_0$ we also get

$$c_0 |\uparrow\rangle = c_0^2 |\downarrow\rangle = 0, \tag{2.83}$$

and we see that the vectors $\{|\downarrow\rangle, |\uparrow\rangle\}$ are the only possible linearly independent states in $\mathcal{H}_G$. This means that the ghost sector of the Hilbert space is two-dimensional and spanned by $|\downarrow\rangle, |\uparrow\rangle$, $\mathcal{H}_G = \text{Span}\{|\downarrow\rangle, |\uparrow\rangle\}$. These states satisfy the orthonormality conditions

$$\langle\uparrow|\uparrow\rangle = \langle\downarrow|\downarrow\rangle = 0, \qquad \langle\downarrow|\uparrow\rangle = \langle\uparrow|\downarrow\rangle = 1. \tag{2.84}$$

In particular the only non-vanishing inner product is given by

$$\langle\downarrow|c_0|\downarrow\rangle = \langle\uparrow|b_0|\uparrow\rangle = 1. \tag{2.85}$$

In the end we obtain that the entire Hilbert space can be spanned by the set of vectors $\{|0, p\rangle \otimes |\downarrow\rangle, |0, p\rangle \otimes |\uparrow\rangle\} = \{|\downarrow, p\rangle, |\uparrow, p\rangle\}$.

We must now understand what are the on-shell states of the theory by studying the co-homology of the the BRST charge $Q_B$. The form of $Q_B$ can be obtained through the Noether theorem as the conserved charge associated with the global BRST symmetry, and it results to be $Q_B = \frac{1}{2}c_0(p^2 + m^2)$, satisfying the following commutation relations:[9]

$$[Q_B, c_0] = [Q_B, p_\mu] = 0, \qquad [Q_B, b_0] = \frac{1}{2}(p^2 + m^2). \tag{2.86}$$

The physical states must be in the cohomology of $Q_B$ so that at first we must impose

$$0 = 2Q_B |{\downarrow}, p\rangle = (p^2 + m^2)|{\uparrow}, p\rangle \; \rightarrow \; p^2 + m^2 = 0, \tag{2.87}$$

$$0 = 2Q_B |{\uparrow}, p\rangle = c_0^2(p^2 + m^2)|{\downarrow}, p\rangle \; \rightarrow \; \text{identically zero.} \tag{2.88}$$

Therefore the kernel of $Q_B$ can be written as

$$\text{Ker}\, Q_B = \text{Span}\left\{|{\uparrow}, p\rangle\right\} \cup \text{Span}\left\{|{\downarrow}, p\rangle\Big|_{p^2+m^2=0}\right\}. \tag{2.89}$$

At the same time it is important to exclude all the exact states in the kernel of $Q$ to isolate the cohomology. To do so it is more practical to define $\#_c$ and $\#_b$ as the ghost and anti-ghost number operators which respectively count the number of $c$ and $b$ present in each state. The total ghost number will then be defined as $\# = \#_c - \#_b = c_0 b_0$ that acts on the states giving

$$\# |{\downarrow}, p\rangle = 0, \qquad \# |{\uparrow}, p\rangle = 1. \tag{2.90}$$

It is obvious then that if there exists a state $|\Lambda\rangle \; | \; |{\downarrow}, p\rangle = Q_B |\Lambda\rangle$ its ghost number must be

$$\# |{\downarrow}, p\rangle = \# (Q_B |\Lambda\rangle) = \# Q_B + \# |\Lambda\rangle \; \rightarrow \; 0 = 1 + \# |\Lambda\rangle \; \rightarrow \; \# |\Lambda\rangle = -1, \tag{2.91}$$

but in the free particle Hilbert space there is not such state with negative ghost number, because the vacuum state is annihilated by the anti-ghost $b_0$. It turns out then that all the closed states at ghost number zero $|{\downarrow}, p\rangle$ are inside the cohomology of $Q$. On the contrary we known that

$$2Q |{\downarrow}, p\rangle = (p^2 + m^2)|{\uparrow}, p\rangle, \tag{2.92}$$

where $(p^2 + m^2)$ is a c-number that, if non-zero, can be inverted, thus allowing us to find the trivializing vector of $|{\uparrow}, p\rangle$:

$$|{\uparrow}, p\rangle = Q\frac{2}{p^2 + m^2}|{\downarrow}, p\rangle, \quad (p^2 + m^2 \neq 0). \tag{2.93}$$

It turns out in the end that all the $|{\uparrow}, p\rangle$ states are not-exact if and only if $p^2 + m^2 = 0$ and the total cohomology of the BRST charge is given by

$$\text{H}(Q_B) = \{|{\downarrow}, p\rangle \oplus |{\uparrow}, p\rangle\}_{p^2+m^2=0}. \tag{2.94}$$

If we then consider the most generic field at ghost number zero, namely

$$|\psi\rangle = \int d^D p\, \psi(p)|{\downarrow}, p\rangle, \tag{2.95}$$

the physicality condition is equivalent to

$$Q |\psi\rangle = 0 \; \rightarrow \; (p^2 + m^2)\psi(p) = 0. \tag{2.96}$$

---

[9]One can easily check that, on the equations of motion, we have $\delta_B \phi = i[\phi, Q_B]$, for $\phi = (X^\mu, c, b)$.

This is the same result we found using the old covariant quantization, where we have already observed that if we change $\psi(p)$ to position space, using the Fourier transform, we get that the "wave function" in position space satisfies the Klein-Gordon equation

$$(\Box - m^2)\tilde{\psi}(x) = 0. \tag{2.97}$$

We can ask what is the role of the states at ghost number 1 (build on $|\uparrow\rangle$), where the cohomology of $Q$ is isomorphic to the one at ghost number 0 (build on $|\downarrow\rangle$). The role of this sector is to provide the full Hilbert space (and hence the cohomology) with a non-degenerate inner product, since ghost number zero states would have identically vanishing inner product with themselves because of (2.84).

Having a non-degenerate inner product is very important: a consequence of this is that the equation $Q|\psi\rangle = 0$ can be obtained as a proper equation of motion by extremizing an action principle of the form

$$S[\psi] = \frac{1}{2} \langle\psi|Q|\psi\rangle, \tag{2.98}$$

which is in this case equivalent to

$$
\begin{aligned}
S[\psi] &= \frac{1}{2}\int d^d p_1 d^d p_2 \, \langle\downarrow,p_1|Q|\downarrow,p_2\rangle\, \psi^\dagger(p_1)\psi(p_2) \\
&= \frac{1}{2}\int d^d p_1 d^d p_2 \, \psi^\dagger(p_1)\langle\downarrow,p_1|\uparrow,p_2\rangle(p_2^2 + m^2)\psi(p_2) \\
&= \frac{1}{2}\int d^d p \, \psi^\dagger(p)(p^2 + m^2)\psi(p) \\
&= \frac{1}{2}\int d^d x \, \psi^\dagger(x)(-\Box + m^2)\psi(x).
\end{aligned}
\tag{2.99}
$$

Notice indeed that this is a genuine space-time action for a (complex) scalar field! This closes our long journey which started from the 1-dimensional field theory of the world-line and, through its BRST quantization, finally lead us to the desired Klein-Gordon action in space-time. A similar construction is available for the string and goes under the name of *String Field Theory*.

### 2.3.1 Practical tools for building the BRST charge

The construction of the BRST charge for the free particle has been quite long but we can notice that, at the end of the day, it boiled down to the simple structure

$$Q_B \sim c \cdot (\text{constraint}) = c_0(p^2 + m^2), \tag{2.100}$$

where the constraint is $p^2 + m^2 = 0$, which is the missing equation of motion for the einbein $e(\tau)$. In fact this structure is more general and it easily generalizes to systems with multiple (compatible) constraints. As we will see, the string falls nicely inside this scheme.

Let us suppose to have a theory equipped with a series of constraints $\mathcal{F}_i$, expressed as operators satisfying a Lie Algebra

$$\left[\mathcal{F}_j, \mathcal{F}_k\right] = \gamma^i{}_{jk}\mathcal{F}_i. \tag{2.101}$$

It turns out that there exists a canonical way to construct the BRST charge directly using the Lie algebra generators $\mathcal{F}_i$. As a starting point, to every constraint $\mathcal{F}_j$ we will associate a pair of ghosts and anti-ghosts $c^j$ and $b_j$. The ghosts will obey

$$[b_i, c^j] = \delta_i^j. \tag{2.102}$$

Together with the structure constants $\gamma^i{}_{jk}$ we can then define new Lie algebra generators acting on the ghost sector as

$$\tilde{\mathcal{F}}_i = -\gamma_{ij}{}^k c^j b_k \,, \tag{2.103}$$

which satisfy the following properties:

$$[\tilde{\mathcal{F}}_i, c^l] = -\gamma_{ij}{}^k[c^j b_k, c^l] = -\gamma_{ij}{}^k c^j[b_k, c^l] = -\gamma_{ij}{}^k c^j \delta_k^l = -\gamma_{ij}{}^l c^j \,, \tag{2.104}$$

$$[\tilde{\mathcal{F}}_i, b_l] = -\gamma_{ij}{}^k[c^j b_k, b_l] = \gamma_{il}{}^k b_k \,. \tag{2.105}$$

Using these two relations we can also prove that $\tilde{\mathcal{F}}_i$ satisfy the same Lie algebra of $\mathcal{F}_i$, namely

$$[\tilde{\mathcal{F}}_i, \tilde{\mathcal{F}}_j] = \gamma_{ij}{}^k \tilde{\mathcal{F}}_k \,. \tag{2.106}$$

This can be proven by direct evaluation:

$$\begin{aligned}
[\tilde{\mathcal{F}}_i, \tilde{\mathcal{F}}_j] &= [\tilde{\mathcal{F}}_i, -\gamma_{jk}{}^l c^k b_l] = -\gamma_{jk}{}^l \left([\tilde{\mathcal{F}}_i, c^k]b_l + c^k[\tilde{\mathcal{F}}_i, b_l]\right) \\
&= -\gamma_{jk}{}^l \left(-\gamma_{im}{}^k c^m b_l + \gamma_{il}{}^m c^k b_m\right) \\
&= \gamma_{jk}{}^l \gamma_{im}{}^k c^m b_l - \gamma_{jk}{}^l \gamma_{il}{}^m c^k b_m \\
&= \left(\gamma_{jk}{}^l \gamma_{im}{}^k - \gamma_{jk}{}^l \gamma_{jm}{}^k\right) c^m b_l \\
&= -\gamma_{ij}{}^k \gamma_{km}{}^l c^m b_l = \gamma_{ij}{}^k \tilde{\mathcal{F}}_k \,,
\end{aligned} \tag{2.107}$$

where in the last steps we have used the fact that the structure constants satisfy the Jacobi identity:

$$\gamma_{jk}{}^l \gamma_{im}^k + \gamma_{ik}{}^l \gamma_{mj}^k + \gamma_{mk}{}^l \gamma_{ji}^k = 0 \,. \tag{2.108}$$

Using the definitions of $\mathcal{F}_i$ and $\tilde{\mathcal{F}}_i$ we can finally identify the BRST charge:

$$Q = c^i \mathcal{F}_i + \frac{1}{2} c^i \tilde{\mathcal{F}}_i = c^i \mathcal{F}_i - \frac{1}{2} \gamma_{ij}{}^k c^i c^j b_k \,. \tag{2.109}$$

If we apply this expression to the ghost fields we obtain

$$[Q, c^l] = \frac{1}{2} c^i [\tilde{\mathcal{F}}_i, c^l] = -\frac{1}{2} \gamma_{ij}{}^l c^i c^j \,, \tag{2.110}$$

$$\begin{aligned}
[Q, b_l] &= \mathcal{F}_i[c^i, b_l] + \frac{1}{2} c^i[\tilde{\mathcal{F}}_i, b_l] + \frac{1}{2}[c^i, b_l]\tilde{\mathcal{F}}_i \\
&= \mathcal{F}_l + \frac{1}{2} \gamma_{li}{}^k c^i b_k + \frac{1}{2} \tilde{\mathcal{F}}_l = \mathcal{F}_l + \tilde{\mathcal{F}}_l = \mathcal{F}_l^{\text{tot}} \,.
\end{aligned} \tag{2.111}$$

Moreover $Q$ is nilpotent:

$$\begin{aligned}
Q^2 &= \frac{1}{2}[Q, Q] = \frac{1}{2}\left[c^i \mathcal{F}_i + \frac{1}{2} c^n \tilde{\mathcal{F}}_n, c^j \mathcal{F}_j + \frac{1}{2} c^m \tilde{\mathcal{F}}_m\right] \\
&= \frac{1}{2}[c^i \mathcal{F}_i, c^j \mathcal{F}_j] + \frac{1}{4}[c^i \mathcal{F}_i, c^m \tilde{\mathcal{F}}_m] + \frac{1}{4}[c^n \tilde{\mathcal{F}}_n, c^j \mathcal{F}_j] + \frac{1}{8}[c^n \tilde{\mathcal{F}}_n, c^m \tilde{\mathcal{F}}_m] \\
&= \frac{1}{2} c^i c^j [\mathcal{F}_i, \mathcal{F}_j] + \frac{1}{2} c^n [\tilde{\mathcal{F}}_n, c^i]\mathcal{F}_i + \frac{1}{8} c^n c^m [\tilde{\mathcal{F}}_n, \tilde{\mathcal{F}}_m] \\
&\quad + \frac{1}{8} c^n[\tilde{\mathcal{F}}_n, c^m]\tilde{\mathcal{F}}_m + \frac{1}{8} c^m[c^n, \tilde{\mathcal{F}}_m]\tilde{\mathcal{F}}_n \\
&= \frac{1}{2} c^i c^j \gamma_{ij}{}^k \mathcal{F}_k - \frac{1}{2} c^n \gamma_{nj}{}^i c^j \mathcal{F}_i + \frac{1}{8} c^n c^m \gamma_{nm}{}^k \tilde{\mathcal{F}}_k \\
&= \frac{1}{8} \gamma_{nm}{}^k \gamma_{ki}{}^j c^n c^m c^i b_j = -\frac{1}{8} \gamma_{[nm}{}^k \gamma_{i]k}{}^j c^n c^m c^i b_j = 0 \,,
\end{aligned} \tag{2.112}$$

where in the last step we have used the Grassmanality of the ghost fields to reconstruct the Jacobi identity.

In the following we will use this construction to directly obtain the BRST charge of string theory.

As a final remark, you may be curious to know whether there is some relation between the (infinite dimensional) structure constants of the gauge group we introduced in (2.34) and the structure constants of the constraints (2.101). In fact, it turns out that in general the gauge fixing procedure which gives rise to the BRST charge leaves some unfixed gauge symmetry. In the case of the particle, for example, these unfixed transformations were constant translations in $\tau$ (as can be readily seen from the gauge fixed matter action (2.17)) which are generated by the Hamiltonian $H = p^2 + m^2$. This unfixed generator is then treated as a constraint. This turns out to be general: the constraints that enter in the BRST charge correspond to the generators of the gauge transformations which are not fixed by the gauge fixing condition. We will find this understanding very useful for the string.

# 3 Free relativistic string

## 3.1 Classical string

Let us start by considering a one-dimensional object, namely a *string*. It can be either open (parametrized by $\sigma \in [0, \pi]$) or closed (parametrized by $\sigma \in [0, 2\pi]$).

As it moves through the target space (i.e. the spacetime), a string spans a two-dimensional surface, called *worldsheet* (WS), that is an open sheet or a closed tube depending on the string being open or closed (see fig. 3.2).

The worldsheet is described by the embedding coordinates $X^\mu(\tau, \sigma)$, where $\tau$ is parametrized time and $\sigma$ is a parametrization of the string itself

$$X^\mu : \quad (\tau, \sigma) \quad \longrightarrow \quad X^\mu(\tau, \sigma) \in \mathbb{R}^{1, D-1} . \tag{3.1}$$

Here $\mu$ is a Lorentz spacetime index $\mu = 0, 1, \ldots, D-1$ (where $D$ is the dimension of the Minkowskian target space). Therefore there are $D$ $X^\mu$ embedding maps, which will be interpreted as $D$ scalar fields in the two dimensions spanned by $(\tau, \sigma)$.

Notice that, in the case of a closed string, the embedding maps have the periodicity condition $X^\mu(\tau, \sigma + 2\pi) = X^\mu(\tau, \sigma)$.

### 3.1.1 Nambu-Goto and Polyakov actions

The dynamics of a string can be described by the *Nambu-Goto action*:

$$S_{\mathrm{NG}}[X] = -T \int_{\mathrm{WS}} dA, \tag{3.2}$$

where $T$ is the string tension and $dA$ is the volume form on the worldsheet. The string's tension $T$ is canonically written as

$$\boxed{T = \frac{1}{2\pi\alpha'},} \tag{3.3}$$

where $[\alpha'] = L^2$ gives the string scale.

Let us now write $\{\tau, \sigma\} = \sigma^\alpha$ (with $\alpha = 0, 1$) and let us denote with $G_{\alpha\beta}$ the induced metric on the worldsheet (where $\alpha, \beta$ are worldsheet indices):

$$G_{\alpha\beta} = \eta_{\mu\nu} \frac{\partial X^\mu}{\partial \sigma^\alpha} \frac{\partial X^\nu}{\partial \sigma^\beta} = \begin{pmatrix} G_{\tau\tau} & G_{\tau\sigma} \\ G_{\sigma\tau} & G_{\sigma\sigma} \end{pmatrix}, \tag{3.4}$$

$$G = \det(G_{\alpha\beta}) = G_{\tau\tau} G_{\sigma\sigma} - (G_{\tau\sigma})^2 . \tag{3.5}$$

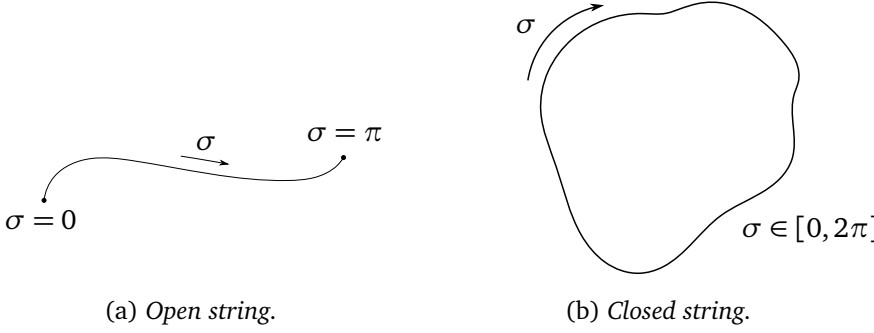

(a) *Open string.*      (b) *Closed string.*

Figure 3.1: Parametrized open and closed strings.

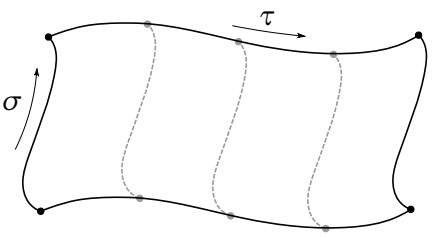
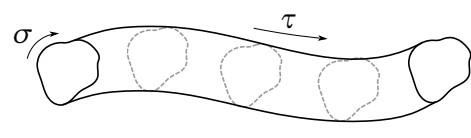

(a) *Open string worldsheet.*          (b) *Closed string worldsheet.*

Figure 3.2: Worldsheet of open and closed strings.

This allows to explicitly write the volume form as

$$dA = \sqrt{-G}\, d^2\sigma\,. \tag{3.6}$$

The Nambu-Goto action can then be rewritten as

$$S_{\text{NG}}[X] = -T \int_{\text{WS}} d^2\sigma \sqrt{-G} = -T \int_{\text{WS}} d^2\sigma \sqrt{-\dot{X}^2 (X')^2 + \left(\dot{X} \cdot X'\right)^2}\,, \tag{3.7}$$

where we used the convention

$$\cdot \equiv \frac{\partial}{\partial \tau}\,, \qquad ' \equiv \frac{\partial}{\partial \sigma}\,. \tag{3.8}$$

As prescribed by the principle of least action, a classical solution of the EOM is a minimal area surface, the higher dimensional generalization of a geodesic. Moreover, $S_{\text{NG}}$ is invariant under reparametrization of $\sigma$. Just as with the particle, this action is not the best starting point for quantization.

To ease the process of quantization, we can write a classically equivalent action which explicitly contains a *dynamical* metric tensor in 2 dimensions $h_{\alpha\beta}(\tau, \sigma)$. This gives the *Polyakov action*[10]

$$S_{\text{P}}[X, h] = -\frac{1}{4\pi\alpha'} \int_{\text{WS}} d^2\sigma \sqrt{-h}\, h^{\alpha\beta} \partial_\alpha X^\mu \partial_\beta X^\nu \eta_{\mu\nu}\,, \tag{3.9}$$

where $h = \det\left(h_{\alpha\beta}\right)$.

### 3.1.2 Equations of motion

Let us first variate $S_{\text{P}}$ with respect to $h$:

$$\delta_h S_{\text{P}} = -\frac{1}{4\pi\alpha'} \int_{\text{WS}} d^2\sigma \left[\delta_h\left(\sqrt{-h}\right) h^{\alpha\beta} \partial_\alpha X^\mu \partial_\beta X^\nu \eta_{\mu\nu} + \sqrt{-h}\, \delta_h\left(h^{\alpha\beta}\right) \partial_\alpha X^\mu \partial_\beta X^\nu \eta_{\mu\nu}\right]$$

$$= -\frac{1}{4\pi\alpha'} \int_{\text{WS}} d^2\sigma \sqrt{-h}\, \delta_h h^{\alpha\beta} \left[-\frac{1}{2} h_{\alpha\beta} \partial_\gamma X^\mu \partial^\gamma X_\mu + \partial_\alpha X^\mu \partial_\beta X_\mu\right]$$

$$= \frac{1}{4\pi} \int_{\text{WS}} d^2\sigma \sqrt{-h}\, \delta_h h^{\alpha\beta} T_{\alpha\beta}\,, \tag{3.10}$$

---

[10]This action was originally written down independently by Deser-Zumino and by Brink-Di Vecchia, but it was used as a path integral by Polyakov.

where we used

$$h^{\alpha\beta}h_{\alpha\beta} = 2, \tag{3.11}$$

$$\delta_h\left(h^{\alpha\beta}h_{\alpha\beta}\right) = 0 = \delta_h\left(h^{\alpha\beta}\right)h_{\alpha\beta} + h^{\alpha\beta}\delta_h\left(h_{\alpha\beta}\right)$$
$$\implies \delta_h\left(h^{\alpha\beta}\right)h_{\alpha\beta} = -h^{\alpha\beta}\delta_h\left(h_{\alpha\beta}\right), \tag{3.12}$$

$$\delta_h h = \delta_h(\det\left(h_{\alpha\beta}\right)) = \delta_h\left(e^{\mathrm{Tr}\{\log\left(h_{\alpha\beta}\right)\}}\right) = \det\left(h_{\alpha\beta}\right)\mathrm{Tr}\left\{\delta_h(\log\left(h_{\alpha\beta}\right))\right\}$$
$$= \det\left(h_{\alpha\beta}\right)\mathrm{Tr}\left\{\frac{\delta_h(h_{\alpha\beta})}{\det\left(h_{\alpha\beta}\right)}\right\} = hh^{\alpha\beta}\delta_h(h_{\alpha\beta})$$
$$= -hh_{\alpha\beta}\delta_h(h^{\alpha\beta}), \tag{3.13}$$

and where we defined the *stress-energy tensor* $T_{\alpha\beta}$ as

$$\alpha' T_{\alpha\beta} = \frac{1}{2}h_{\alpha\beta}\partial_\gamma X^\mu\partial^\gamma X_\mu - \partial_\alpha X^\mu\partial_\beta X_\mu$$
$$= \frac{1}{2}h_{\alpha\beta}h^{\gamma\delta}G_{\gamma\delta} - G_{\alpha\beta}. \tag{3.14}$$

Therefore, if we ask for $\delta_h S_P = 0$, we find the $h$-EOM

$$\boxed{T_{\alpha\beta} = 0.} \tag{3.15}$$

This equation implies that

$$G_{\alpha\beta} = \left(\frac{1}{2}h^{\gamma\delta}G_{\gamma\delta}\right)h_{\alpha\beta}, \tag{3.16}$$

meaning that the induced metric $G_{\alpha\beta}$ is proportional to the free metric $h_{\alpha\beta}$ through the scale parameter $\frac{1}{2}h^{\gamma\delta}G_{\gamma\delta}$. Using the $h$-EOM (3.15) we then get

$$h_{\alpha\beta} = \left(\frac{1}{2}h^{\gamma\delta}G_{\gamma\delta}\right)^{-1}G_{\alpha\beta}$$
$$\implies h = \left(\frac{1}{2}h^{\gamma\delta}G_{\gamma\delta}\right)^{-2}\det\left(G_{\alpha\beta}\right) = \left(\frac{1}{2}h^{\gamma\delta}G_{\gamma\delta}\right)^{-2}G$$
$$\implies \sqrt{-h} = \left(\frac{1}{2}h^{\alpha\beta}G_{\alpha\beta}\right)^{-1}\sqrt{-G}. \tag{3.17}$$

Therefore, thanks to $h$-EOM (3.15), we can see that Polyakov and Nambu-Goto actions are the same:

$$S_P = -\frac{1}{4\pi\alpha'}\int_{WS}d^2\sigma\sqrt{-G}\left(\frac{1}{2}h^{\gamma\delta}G_{\gamma\delta}\right)^{-1}h^{\alpha\beta}G_{\alpha\beta}$$
$$= -\frac{1}{2\pi\alpha'}\int_{WS}d^2\sigma\sqrt{-G}$$
$$= S_{NG}. \tag{3.18}$$

Moreover, recalling the definition (3.14), we can see that

$$\alpha' T^\alpha{}_\alpha = -(\partial X\cdot\partial X) + \frac{1}{2}h^\alpha_\alpha(\partial X\cdot\partial X)$$
$$= -(\partial X\cdot\partial X) + \frac{1}{2}2(\partial X\cdot\partial X) = 0. \tag{3.19}$$

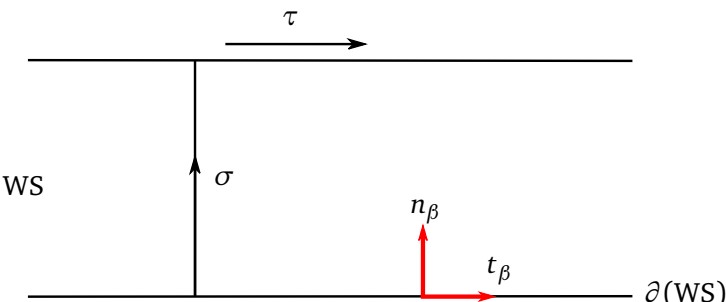

Figure 3.3: The normal and the tangent vector with respect to the boundary $\partial(\mathrm{WS})$ of the open string worldsheet.

Therefore, thanks to the fact that we are in two dimensions, we find that the stress-energy tensor is traceless:

$$\boxed{T^{\alpha}{}_{\alpha} = 0.} \tag{3.20}$$

Let us now vary $S_{\mathrm{P}}$ with respect to $X$:

$$
\begin{aligned}
\delta_X S_{\mathrm{P}} &= -\frac{1}{2\pi\alpha'} \int_{\mathrm{WS}} d^2\sigma \, \sqrt{-h} \, h^{\alpha\beta} \partial_\alpha \delta_X X^\mu \partial_\beta X_\mu \\
&\overset{\mathrm{IBP}}{=} \frac{1}{2\pi\alpha'} \int_{\mathrm{WS}} d^2\sigma \left[ \delta_X X^\mu \partial_\alpha \left( \sqrt{-h} \, h^{\alpha\beta} \partial_\beta X_\mu \right) - \partial_\alpha \left( \sqrt{-h} \, h^{\alpha\beta} \delta_X X^\mu \partial_\beta X_\mu \right) \right],
\end{aligned} \tag{3.21}
$$

where in the last step we performed an integration by parts (IBP).

If we then ask for $\delta_X S_{\mathrm{P}} = 0$, the resulting $X$-EOMs are

$$
\begin{cases}
\partial_\alpha \left( \sqrt{-h} \, h^{\alpha\beta} \partial_\beta X_\mu \right) = 0, \\
\left[ \sqrt{-h} \, h^{\alpha\beta} \delta_X X_\mu \partial_\beta X_\mu \right]_{\partial(\mathrm{WS})} = 0,
\end{cases} \tag{3.22}
$$

where $\partial(\mathrm{WS})$ is the worldsheet boundary, which is only present in the case of open strings.

We shall now briefly focus on open strings. Since

$$\int_{\mathrm{WS}} d^2\sigma = \int_0^\pi d\sigma \int_{-\infty}^{+\infty} d\tau, \tag{3.23}$$

we can write the boundary term (3.22) as

$$
\begin{aligned}
\int_{-\infty}^{+\infty} d\tau \int_0^\pi d\sigma \, \partial_\sigma \left[ \delta X^\mu \sqrt{-h} \, h^{\sigma\beta} \partial_\beta X_\mu \right] \\
= \int_{-\infty}^{+\infty} d\tau \, \sqrt{-h} \, h^{\sigma\beta} \left[ \delta X^\mu \partial_\beta X_\mu \right]_{\sigma=0}^{\sigma=\pi} = 0,
\end{aligned} \tag{3.24}
$$

where $h^{\sigma\beta}$ is a vector, being $\sigma$ a fixed index.

Let us call $n$ the vector normal to $\partial(\mathrm{WS})$ and $t$ the vector tangent to $\partial(\mathrm{WS})$ (see fig. 3.3)

$$n^\beta = h^{\sigma\beta}, \tag{3.25}$$

$$t_\beta = h_{\tau\beta}, \tag{3.26}$$

$$t_\beta n^\beta = \delta^\sigma_\tau = 0. \tag{3.27}$$

Then we end up with the following $X$-EOMs

$$
\begin{cases}
\partial_\alpha \left( \sqrt{-h}\, h^{\alpha\beta} \partial_\beta X_\mu \right) = 0 & \text{(WS bulk eq.),} \\
\delta X^\mu \left( n^\alpha \partial_\alpha X_\mu \right)\big|_{\partial(\text{WS})} = 0 & \text{(WS boundary eq.),}
\end{cases}
\tag{3.28}
$$

where the worldsheet boundary equation is for open strings only.

### 3.1.3   Internal symmetries

The Polyakov action $S_P[h, X]$ is invariant under the following local transformations:

- diffeomorphisms, namely reparametrizations of the type $\sigma' = \sigma'(\sigma)$ acting as follows:

$$
\begin{cases}
X'(\sigma') = X(\sigma), \\
\frac{\partial \sigma'^\gamma}{\partial \sigma^\alpha} \frac{\partial \sigma'^\delta}{\partial \sigma^\beta} h'_{\gamma\delta}(\sigma') = h_{\alpha\beta}(\sigma).
\end{cases}
\tag{3.29}
$$

- Weyl symmetries, namely metric rescalings that do not affect the $X$ field:

$$
h'_{\alpha\beta}(\sigma) = e^{2\omega(\sigma)} h_{\alpha\beta}(\sigma).
\tag{3.30}
$$

The diff-symmetry is obvious from general covariance. Let us instead check the Weyl symmetry. We have that

$$
(h^{\alpha\beta})' = e^{-2\omega(\sigma)} h^{\alpha\beta}(\sigma),
\tag{3.31}
$$

$$
h' = \det\left( h'_{\alpha\beta} \right) = e^{4\omega(\sigma)} h \quad \Longrightarrow \quad \sqrt{-h'} = e^{2\omega(\sigma)} \sqrt{-h},
\tag{3.32}
$$

and hence

$$
\sqrt{-h'}\, h'^{\alpha\beta} = \sqrt{-h}\, h^{\alpha\beta}.
\tag{3.33}
$$

Therefore we found that, being $\sqrt{-h}\, h^{\alpha\beta}$ Weyl invariant, such a transformation is a symmetry for $S_P$, as claimed before.

Moreover, if we write

$$
h'_{\alpha\beta} = e^{2\omega} h_{\alpha\beta} = (1 + 2\omega + \mathcal{O}(\omega^2)) h_{\alpha\beta},
\tag{3.34}
$$

we can express the infinitesimal Weyl transformation as

$$
\delta_\omega h_{\alpha\beta} = 2\omega h_{\alpha\beta},
\tag{3.35}
$$

and then we can write

$$
\delta_\omega S_P \propto \int_{\text{WS}} d^2\sigma \sqrt{-h}\, \delta_\omega h^{\alpha\beta} T_{\alpha\beta} = 0
$$
$$
\Longrightarrow h^{\alpha\beta} T_{\alpha\beta} = T^\alpha_\alpha = 0.
\tag{3.36}
$$

The energy-momentum tensor is traceless as a consequence of Weyl invariance.

Besides the above gauge symmetries there is also a global Poincaré symmetry in $D$ dimensions

$$
\begin{cases}
X'^\mu = \Lambda^\mu{}_\nu X^\nu + a^\nu, \\
h'_{\alpha\beta} = h_{\alpha\beta}.
\end{cases}
\tag{3.37}
$$

## 3.2 Old covariant quantization

We now want to quantize the free relativistic string. To do this we have to gauge fix the Weyl+Diff gauge redundancy we have just described. This can be done by introducing the convenient *conformal gauge* fixing, corresponding to the choice of a fiducial metric

$$\hat{h}_{\alpha\beta} = e^{-\phi}\eta_{\alpha\beta}, \tag{3.38}$$

$$\hat{h}^{\alpha\beta} = e^{\phi}\eta^{\alpha\beta}, \tag{3.39}$$

where $\phi = \phi(\sigma)$ is a worldsheet dependent scale factor. In this gauge we have that

$$\sqrt{-\hat{h}} = e^{-\phi}\sqrt{-\eta} = e^{-\phi}, \tag{3.40}$$

and therefore

$$\hat{h}^{\alpha\beta}\sqrt{-\hat{h}} = \eta^{\alpha\beta}. \tag{3.41}$$

Notice that $\phi$ decouples[11] from the object $\hat{h}^{\alpha\beta}\sqrt{-\hat{h}}$. Therefore the gauge fixed action is

$$S[\hat{h},\hat{X}] = -\frac{1}{4\pi\alpha'}\int_{\text{WS}} d^2\sigma\, \eta^{\alpha\beta}\partial_\alpha X^\mu \partial_\beta X_\mu. \tag{3.42}$$

It is now convenient to change to the *lightcone coordinates*

$$\boxed{\sigma^\pm = \tau \pm \sigma.} \tag{3.43}$$

They allow us to write the integration measure as

$$d^2\sigma = \frac{1}{2}d\sigma^+ d\sigma^-, \tag{3.44}$$

and

$$\tau = \frac{\sigma^+ + \sigma^-}{2}, \qquad \sigma = \frac{\sigma^+ - \sigma^-}{2}. \tag{3.45}$$

We can thus write

$$\eta_{++} = \eta_{--} = 0, \qquad \eta^{++} = \eta^{--} = 0, \tag{3.46a}$$

$$\eta_{+-} = \eta_{-+} = -\frac{1}{2}, \qquad \eta^{+-} = \eta^{-+} = -2, \tag{3.46b}$$

namely

$$\eta = \begin{pmatrix} 0 & -\frac{1}{2} \\ -\frac{1}{2} & 0 \end{pmatrix}, \qquad \eta^{-1} = \begin{pmatrix} 0 & -2 \\ -2 & 0 \end{pmatrix}. \tag{3.47}$$

Using these coordinates, the action reads

$$S = \frac{1}{2\pi\alpha'}\int_{\text{WS}} d\sigma^+ d\sigma^-\, \partial_+ X^\mu \partial_- X_\mu. \tag{3.48}$$

We may now pause for a moment and observe that, even after the gauge fixing, this action exibits a residual gauge freedom under *conformal diffeomorphisms*, defined as

$$\sigma'^+ = f(\sigma^+) = f^+, \tag{3.49a}$$

$$\sigma'^- = g(\sigma^-) = g^-. \tag{3.49b}$$

---

[11]At the quantum level it can be shown that this holds only for $D = 26$.

Under such a transformation, the metric tensor is changed by a scale factor:

$$-d\sigma^{'-}d\sigma^{'+} = -(\partial_+ f_+ \partial_- g_-)\,df^+ dg^-\,. \tag{3.50}$$

Therefore the metric can be recast to its original form using a Weyl transformation. This residual gauge symmetry os called conformal invariance and it will be central in the following developments.

The $X$-EOMs in the conformal gauge read

$$\begin{cases} \partial_+ \partial_- X^\mu(\sigma^+, \sigma^-) &= 0 \qquad \text{(WS bulk eq.),} \\ \delta X_\mu \partial_\sigma X^\mu \big|_{\sigma=0,\pi} &= 0 \qquad \text{(WS boundary eq.).} \end{cases} \tag{3.51}$$

At the same time it is important to consider the energy momentum tensor (which is set to zero by the missing equation of motion for the metric, which is no more obtainable by varying the gauge-fixed action).

Using the convention

$$T_{\alpha\beta} = \frac{4\pi}{\sqrt{-h}}\frac{\delta S}{\delta h^{\alpha\beta}} = \frac{1}{2\alpha'}h_{\alpha\beta}\partial_\gamma X^\mu \partial^\gamma X_\mu - \frac{1}{\alpha'}\partial_\alpha X^\mu \partial_\beta X_\mu\,, \tag{3.52}$$

we have

$$T_{++} = -\frac{1}{\alpha'}\left(\partial_+ X^\mu \partial_+ X_\mu\right), \tag{3.53a}$$

$$T_{--} = -\frac{1}{\alpha'}\left(\partial_- X^\mu \partial_- X_\mu\right), \tag{3.53b}$$

$$\alpha' T_{+-} = \alpha' T_{-+} = -\partial_+ X^\mu \partial_- X^\mu + \frac{1}{2}2\left(\partial_+ X^\mu \partial_- X^\mu\right) = 0\,. \tag{3.53c}$$

Notice that that (upon use of the $X$-eoms) the energy momentum tensor is conserved:

$$\partial^\alpha T_{\alpha\beta} = 0 \iff \begin{cases} \partial_+ T_{--} = 0\,, \\ \partial_- T_{++} = 0\,. \end{cases} \tag{3.54}$$

This implies that

$$\begin{cases} T_{++} = T_{++}(\sigma^+)\,, \\ T_{--} = T_{--}(\sigma^-)\,. \end{cases} \tag{3.55}$$

We are now ready to quantize the string in the "old" manner, namely through *old covariant quantization* (OCQ).

### 3.2.1 OCQ for the closed string

#### 3.2.1.1 Classical solution

Let us focus first on the closed string case. The EOM will be the bulk equation

$$\partial_+ \partial_- X^\mu(\sigma^+, \sigma^-) = 0\,. \tag{3.56}$$

This is a wave equation in two dimensions, whose generic solution is

$$X^\mu(\sigma^+, \sigma^-) = X_L^\mu(\sigma^+) + X_R^\mu(\sigma^-)\,, \tag{3.57}$$

where $X_L^\mu$ is the left-moving wave, while $X_R^\mu$ is the right-moving one.

Recalling that the solution has to be periodic in $\sigma \rightarrow \sigma + 2\pi$ (namely $X^\mu(\tau, \sigma + 2\pi) = X^\mu(\tau, \sigma)$) we can write it as

$$X_L^\mu(\sigma^+) = \frac{1}{2}\left(X_0^\mu + c^\mu\right) + \frac{\alpha'}{2}P^\mu\sigma^+ + i\sqrt{\frac{\alpha'}{2}}\sum_{n\neq0}\frac{\alpha_n^\mu}{n}e^{-in\sigma^+}, \tag{3.58a}$$

$$X_R^\mu(\sigma^-) = \frac{1}{2}\left(X_0^\mu - c^\mu\right) + \frac{\alpha'}{2}P^\mu\sigma^- + i\sqrt{\frac{\alpha'}{2}}\sum_{n\neq0}\frac{\tilde{\alpha}_n^\mu}{n}e^{-in\sigma^-}, \tag{3.58b}$$

where $c^\mu$ is a zero-mode freedom which disappears in the sum $X_L + X_R$ to which we are ultimately interested.

Hence the generic solution reads as follows:

$$X^\mu(\sigma^+, \sigma^-) = X_0^\mu + \alpha'P^\mu\tau + i\sqrt{\frac{\alpha'}{2}}\sum_{n\neq0}\frac{1}{n}\left(\alpha_n^\mu e^{-in\sigma^+} + \tilde{\alpha}_n^\mu e^{-in\sigma^-}\right). \tag{3.59}$$

This is the classical closed string solution. It resembles what we already found for the free particle[12] (2.21a), being $\alpha'P^\mu\tau$ the term containing the information on the center of mass (as we will see shortly, $P^\mu$ is indeed the momentum of the center of mass). Moreover, the string solution has the additional term $i\sqrt{\frac{\alpha'}{2}}\sum_{n\neq0}\frac{1}{n}\left(\alpha_n^\mu e^{-in\sigma^+} + \tilde{\alpha}_n^\mu e^{-in\sigma^-}\right)$, which describes the internal oscillatory dynamics of the string. The $i$ is needed in order to implement the $X^\mu$ reality condition:

$$X^\mu = (X^\mu)^* \iff \overset{(\sim)}{\alpha}_{-n} = (\overset{(\sim)}{\alpha}_n)^*. \tag{3.60}$$

Let us now evaluate the momentum density

$$\begin{aligned}\mathcal{P}^\mu(\tau, \sigma) &= \frac{\partial\mathcal{L}}{\partial\dot{X}_\mu} = \frac{1}{2\pi\alpha'}\dot{X}^\mu \\ &= \frac{P^\mu}{2\pi} + \frac{1}{2\pi\sqrt{2\alpha'}}\sum_{n\neq0}\left(\alpha_n^\mu e^{-in(\tau+\sigma)} + \tilde{\alpha}_n^\mu e^{-in(\tau-\sigma)}\right),\end{aligned} \tag{3.61}$$

where we used the $X$-EOM. If we integrate this density over the string extension

$$\int_0^{2\pi} d\sigma\, \mathcal{P}^\mu = P^\mu, \tag{3.62}$$

we see that indeed $P^\mu$ is the momentum of the center of mass.

### 3.2.1.2 Quantization

In canonical quantization we now have to impose the equal time commutation relation

$$\left[X^\mu(\tau, \sigma), \mathcal{P}^\nu(\tau, \sigma')\right] = i\eta^{\mu\nu}\delta(\sigma - \sigma'), \tag{3.63}$$

---

[12]Here, for the string, we take the worldsheet time $\tau$ to be a-dimensional, while for the particle we were implicitly assuming that $\tau$ had dimensions of (space-time) time$^2$.

on the solutions (3.59), (3.61). At the level of oscillators this means

$$\left[\alpha_n^\mu,\alpha_m^\nu\right]=\left[\tilde{\alpha}_n^\mu,\tilde{\alpha}_m^\nu\right]=n\eta^{\mu\nu}\delta_{n+m,0}\,, \tag{3.64a}$$

$$\left[\alpha_n^\mu,\tilde{\alpha}_m^\nu\right]=0\,, \tag{3.64b}$$

$$\left[X_0^\mu,P^\nu\right]=i\eta^{\mu\nu}\,. \tag{3.64c}$$

**Exercise 3.2.1**

Prove this statement, using the formula $\delta(\sigma)=\sum_{n\in\mathbb{Z}}\frac{e^{in\sigma}}{2\pi}$.

If we define

$$a_n^\mu=\frac{1}{\sqrt{n}}\alpha_n^\mu\,,\qquad\text{for }n>0\,, \tag{3.65}$$

$$a_n^{\mu\dagger}=\frac{1}{\sqrt{n}}\alpha_{-n}^\mu\,,\qquad\text{for }n>0\,, \tag{3.66}$$

we then have

$$\left[a_n^\mu,a_m^{\nu\dagger}\right]=\eta^{\mu\nu}\delta_{m,n}\,. \tag{3.67}$$

These are therefore infinitely many harmonic oscillators. The vacuum is defined as

$$\alpha_n|0\rangle=0\,,\qquad\langle0|\alpha_{-n}=0\,,\qquad\forall n>0\,, \tag{3.68}$$

with the reality condition

$$\left(\alpha_n^\mu\right)^\dagger=\alpha_{-n}^\mu\,. \tag{3.69}$$

Taking into account also the center-of-mass Heisenberg algebra $\left[X_0^\mu,P^\nu\right]=i\eta^{\mu\nu}$, the Hilbert space will thus be generated by the vacuum at definite momentum $|0,P\rangle$, on which oscillators act. The generic basis element will thus be

$$\alpha_{-n_1}^{\mu_1}\dots\alpha_{-n_k}^{\mu_k}\tilde{\alpha}_{-m_1}^{\nu_1}\dots\tilde{\alpha}_{-m_k}^{\nu_k}|0,P\rangle\,. \tag{3.70}$$

Thanks to the oscillators, such a state can be in any (possibly reducible) Lorentz representation $(\{\mu_1,\dots,\mu_k\}\{\nu_1,\dots,\nu_k\})$ of integer spin. The momentum is also not constrained in any way. This is the most general off-shell state we can write down. Next, to find the physical states, we have to impose the constraints which are the missing equation for the metric, resulting in the vanishing of the energy-momentum tensor.

### 3.2.1.3 Physical constraints and Virasoro algebra

Now we have to implement the missing equation of motion for $h$ as constraints:

$$T_{++}=-\frac{1}{\alpha'}\partial_+X^\mu\partial_+X_\mu=T_{++}(\sigma^+)\,, \tag{3.71}$$

$$T_{--}=-\frac{1}{\alpha'}\partial_-X^\mu\partial_-X_\mu=T_{--}(\sigma^-)\,. \tag{3.72}$$

Expanding them in harmonics

$$\begin{cases}T_{++}(\sigma^+)=-\sum_{n\in\mathbb{Z}}L_ne^{-in\sigma^+}\,,\\T_{--}(\sigma^-)=-\sum_{n\in\mathbb{Z}}\tilde{L}_ne^{-in\sigma^-}\,,\end{cases} \tag{3.73}$$

we obtain the classical expressions of the so-called *Virasoro operators*

$$L_n = \frac{1}{2}\sum_{k\in\mathbb{Z}}\alpha^\mu_{n-k}\alpha^\nu_k\eta_{\mu\nu}\,, \qquad \widetilde{L}_n = \frac{1}{2}\sum_{k\in\mathbb{Z}}\tilde{\alpha}^\mu_{n-k}\tilde{\alpha}^\nu_k\eta_{\mu\nu}\,, \tag{3.74}$$

where the zero-mode is defined as

$$\alpha^\mu_0 = \tilde{\alpha}^\mu_0 = \sqrt{\frac{\alpha'}{2}}P^\mu \quad \text{(closed string)}. \tag{3.75}$$

**Exercise 3.2.2**

Obtain explicitly the Virasoro operators.

We have now to interepret the above classical expressions as operators. In doing so we have to pay attention that the oscillators $\alpha^\mu_n$ are not always commuting and therefore the previous classical expressions may be ambiguous. In fact closer inspection reveals that *only* $L_0$ suffers from a possible ambiguity:

$$L_{n\neq 0} = \frac{1}{2}\sum_{k\in\mathbb{Z}}\alpha_{n-k}\cdot\alpha_k \quad \text{(non ambiguos)}, \tag{3.76}$$

$$L_0 \sim \frac{1}{2}\sum_{k\in\mathbb{Z}}\alpha_{-k}\cdot\alpha_k \quad \text{(ambiguous!)}, \tag{3.77}$$

because

$$\left[\alpha^\mu_{n-k},\alpha^\nu_k\right] = 0\,, \tag{3.78a}$$

$$\left[\alpha^\mu_{-k},\alpha^\nu_k\right] \neq 0\,. \tag{3.78b}$$

A natural possibility to consider is the oscillator normal ordering $:\cdots:$, where creation operators are placed to the left and annihilation operators to the right:

$$:L_0: = \frac{1}{2}\sum_{k\in\mathbb{Z}}:\alpha_{-k}\cdot\alpha_k: = \frac{1}{2}\alpha_0^2 + \sum_{k\geq 0}\alpha_{-k}\cdot\alpha_k \equiv \hat{L}_0\,. \tag{3.79}$$

Since any different ordering convention will result in a constant shift in $L_0$, for the time being we leave this constant free and fix it a later stage from consistency. This brings to the physical state constraint

$$\begin{cases} \left(\hat{L}_0 - a\right)|\text{phys}\rangle = 0\,, \\ \left(\hat{\widetilde{L}}_0 - \tilde{a}\right)|\text{phys}\rangle = 0\,. \end{cases} \tag{3.80}$$

From now on, if not needed, we will not explicitly write the $\hat{}$ on quantum operators, as well as the normal ordering. Moreover, in the following we will always focus on $L_n$, since the story for the $\widetilde{L}_n$ is the same.

Before discussing how to impose the constraints related to the $L_{n\neq 0}$'s, it is important to study their operator algebra. We may start by observing that

$$\left[L_n,\alpha^\mu_m\right] = -m\alpha^\mu_{n+m}\,. \tag{3.81}$$

**Exercise 3.2.3**

Prove eq. (3.81).

We can then evaluate

$$[L_n, L_m] = \frac{1}{2} \sum_{k \in \mathbb{Z}} [L_n, :\alpha_{m-k} \cdot \alpha_k:]$$

$$= \frac{1}{2} \left( \sum_{k \geq 0} [L_n, \alpha_{m-k} \cdot \alpha_k] + \sum_{k < 0} [L_n, \alpha_k \cdot \alpha_{m-k}] \right)$$

$$= \frac{1}{2} \left( \sum_{k \geq 0} \{ (k-m)\alpha_{n+m-k} \cdot \alpha_k - k\alpha_{m-k} \cdot \alpha_{n+k} \} \right.$$

$$\left. + \sum_{k < 0} \{ -k\alpha_{n+k} \cdot \alpha_{m-k} + (k-m)\alpha_k \cdot \alpha_{n+m-k} \} \right). \qquad (3.82)$$

Let us now choose $n > 0$ for convenience and set $n + k = q$ in the second and third terms. We then have

$$[L_n, L_m] = \frac{1}{2} \left( \sum_{k \geq 0} (k-m)\alpha_{n+m-k} \cdot \alpha_k + \sum_{q \geq n} (n-q)\alpha_{n+m-q} \cdot \alpha_q \right.$$

$$\left. + \sum_{q \leq n-1} (n-q)\alpha_q \cdot \alpha_{n+m-q} + \sum_{k < 0} (k-m)\alpha_k \cdot \alpha_{n+m-k} \right)$$

$$= \frac{1}{2} \left( \sum_{k \geq 0} (n-m)\alpha_{n+m-k} \cdot \alpha_k + \sum_{k=0}^{n-1} (k-n)\alpha_{n+m-k} \cdot \alpha_k \right.$$

$$\left. + \sum_{k \leq -1} (n-m)\alpha_k \cdot \alpha_{n+m-k} + \sum_{k=0}^{n-1} (n-k)\alpha_k \cdot \alpha_{n+m-k} \right). \qquad (3.83)$$

Moreover we can write

$$\sum_{k=0}^{n-1} (n-k)\alpha_k \cdot \alpha_{n+m-k} = \sum_{k=0}^{n-1} (n-k)\left( \alpha_{n+m-k} \cdot \alpha_k - \left[ \alpha_{n+m-k}^\mu, \alpha_k^\nu \right] \eta_{\mu\nu} \right)$$

$$= \sum_{k=0}^{n-1} (n-k)\left( \alpha_{n+m-k} \cdot \alpha_k - (-k)\eta^{\mu\nu}\delta_{n+m,0}\eta_{\mu\nu} \right)$$

$$= \sum_{k=0}^{n-1} (n-k)\left( \alpha_{n+m-k} \cdot \alpha_k + kd\delta_{n+m,0} \right), \qquad (3.84)$$

where $d$ is the target space dimension. Therefore we have

$$[L_n, L_m] = \frac{1}{2} \left( \sum_{k \in \mathbb{Z}} (n-m)\alpha_{n+m-k} \cdot \alpha_k + \sum_{k=0}^{n-1} (n-k)k \, D \, \delta_{n+m,0} \right)$$

$$= (n-m)L_{n+m} + \frac{1}{2} D \sum_{k=0}^{n-1} (n-k)k\delta_{n+m,0}. \qquad (3.85)$$

If we now evaluate the finite sum

$$\sum_{k=0}^{n-1} (n-k)k = \frac{1}{6}n(n^2-1), \qquad (3.86)$$

we find the *quantum* algebra staisfied by the constraints

$$[L_n, L_m] = (n-m)L_{n+m} + \frac{D}{12}n(n^2-1)\delta_{n+m,0}\,, \tag{3.87}$$

which is the famous *Virasoro algebra* with *central charge* $c = D$. The term

$$\frac{c}{12}n(n^2-1)\delta_{n+m,0}$$

is called a *central extension* and, as we have seen, is a (two-dimensional) quantum effect.

We immediately notice that the central extension vanishes when $n \neq -m$.

Moreover, it vanishes also when $n = -1, 0, +1$, which means that $L_{-1}, L_0, L_1$ do not have the central extension. They obey the nice $SL(2, \mathbb{C})$ algebra

$$[L_0, L_1] = -L_1\,, \qquad [L_0, L_{-1}] = +L_{-1}\,, \qquad [L_1, L_{-1}] = 2L_0\,, \tag{3.88}$$

also known as the *Gliozzi algebra* (who 're-discovered' it in the early days of string theory) or the *small conformal algebra*. We recall that $SL(2, \mathbb{C})$ can be put in (almost) one-to-one correspondence with the group of global conformal transformations of the Riemann sphere

$$f(z) = \frac{\alpha z + \beta}{\gamma z + \delta}\,, \tag{3.89}$$

whose coefficients matrix has unit determinant:

$$\det \begin{pmatrix} \alpha & \beta \\ \gamma & \delta \end{pmatrix} = 1\,. \tag{3.90}$$

As we will see, this is no coincidence and it will be better understood when we will study the underlying conformal field theory (CFT) on the worldsheet.

Let us now consider the zero-momentum vacuum:

$$|0, P\rangle\Big|_{P=0} = |0\rangle\,. \tag{3.91}$$

We have that

$$P^\mu |0\rangle = 0\,, \tag{3.92a}$$

$$\alpha_0^\mu |0\rangle = 0\,, \tag{3.92b}$$

$$\tilde{\alpha}_0^\mu |0\rangle = 0\,. \tag{3.92c}$$

Notice that the normal ordering $:\cdots:$ we introduced earlier is well-adapted to $|0\rangle$, even for the $\alpha_0$ oscillator which obviously commutes with everything. We then have

$$L_{+1} |0\rangle = 0\,, \tag{3.93a}$$

$$L_0 |0\rangle = 0\,, \tag{3.93b}$$

$$L_{-1} |0\rangle = 0\,, \tag{3.93c}$$

since

$$L_{+1} |0\rangle = \frac{1}{2}\sum_{k\in\mathbb{Z}} \alpha_{1-k}\cdot\alpha_k |0\rangle = \begin{cases} 0\,, & \text{for } k > 0\text{, since } \alpha_k \text{ annihilates } |0\rangle\,, \\ 0\,, & \text{for } k < 0\text{, since } \alpha_{1-k} \text{ annihilates } |0\rangle\,, \\ 0\,, & \text{for } k = 0\text{, since } \alpha_0\big|_{P=0} = 0\,, \end{cases} \tag{3.94}$$

$$L_0 |0\rangle = \frac{1}{2} \sum_{k \in \mathbb{Z}} \alpha_{-k} \cdot \alpha_k = \left( \frac{1}{2}(\alpha_0)^2 + \frac{1}{2} \sum_{k \neq 0} \alpha_{-k} \cdot \alpha_k \right) |0\rangle$$

$$= \left( \frac{\alpha' P^2}{4} \Big|_{P=0} + \sum_{k>0} \alpha_{-k} \cdot \alpha_k \right) |0\rangle$$

$$= 0 \,, \tag{3.95}$$

$$L_{-1} |0\rangle = \frac{1}{2} \sum_{k \in \mathbb{Z}} \alpha_{-1-k} \cdot \alpha_k |0\rangle = \begin{cases} 0, & \text{for } k > 0, \text{ since } \alpha_{-1-k} \text{ annihilates } |0\rangle, \\ 0, & \text{for } k < 0, \text{ since } \alpha_k \text{ annihilates } |0\rangle, \\ 0, & \text{for } k = 0, \text{ since } \alpha_0 \big|_{P=0} = 0. \end{cases} \tag{3.96}$$

This means that $|0\rangle$ is invariant under the action of $SL(2,\mathbb{C})$ and therefore it is called $SL(2,\mathbb{C})$-*invariant vacuum*. From the point of view of the zero-momentum vacuum $|0\rangle$, the states $|0,P\rangle$ are not vacua, but rather excited states obtained by injecting momentum $P$ into $|0\rangle$ by means of a plane wave operator:

$$e^{iP \cdot \hat{X}_0} |0\rangle = |0,P\rangle \,. \tag{3.97}$$

We have indeed

$$\hat{P}^\mu e^{iP \cdot \hat{X}_0} |0\rangle = 0 + \left[ \hat{P}^\mu, e^{iP \cdot \hat{X}_0} \right] |0\rangle$$

$$= iP_\nu \left[ \hat{P}^\mu, \hat{X}_0^{\ \nu} \right] e^{iP \cdot \hat{X}_0} |0\rangle$$

$$= iP_\nu (-i\eta^{\mu\nu}) e^{iP \cdot \hat{X}_0} |0\rangle$$

$$= P^\mu e^{iP \cdot \hat{X}_0} |0\rangle \,. \tag{3.98}$$

Therefore $|0\rangle$ is the true *unique* vacuum of our matter Hilbert space.

Now we are ready to impose the remaining Virasoro constraints in addition to (3.80). When $n \neq 0$, we would be tempted to define the physical states $|\text{phys}\rangle$ as

$$L_n |\text{phys}\rangle \overset{?}{=} 0, \quad \forall n \neq 0. \tag{3.99}$$

However, because of the Virasoro algebra, this way of imposing the constraints would result in

$$\langle \text{phys} | [L_n, L_m] | \text{phys'} \rangle = 0$$

$$= \langle \text{phys} | \left( (n-m) L_{n+m} + \frac{D}{12} n(n^2-1) \delta_{n+m,0} \right) | \text{phys'} \rangle, \tag{3.100}$$

which would imply

$$\langle \text{phys} | \text{phys'} \rangle = 0. \tag{3.101}$$

Since this is clearly not acceptable, we take the weaker constraint

$$\langle \text{phys} | L_n | \text{phys'} \rangle = 0, \quad \forall n \neq 0, \tag{3.102}$$

which implies (recalling that $L_n^\dagger = L_{-n}$)

$$\boxed{L_n |\text{phys}\rangle = 0, \qquad \langle \text{phys} | L_{-n} = 0, \quad \forall n > 0.} \tag{3.103}$$

The same holds for $\widetilde{L}_n$.

On the other hand, if we consider $n = 0$, the physical constraint we have to consider is the one that we already wrote in (3.80), namely

$$(L_0 - a)\,|\text{phys}\rangle = 0\,, \tag{3.104}$$

where we recalled that $L_0^\dagger = L_0$. We will see that, for a proper critical value of $a$ (and of $D$, the space-time dimensions), these constraints will give rise to a consistent physical spectrum of excitations with well-defined interactions. However to fully understand the mathematical reason behind these critical values we will need the BRST quantization.

### 3.2.1.4 Two anticipations from BRST quantization

At this point it is worth noticing that the quantum system we are studying has two unfixed quantities: the spacetime dimension $D$ and the normal ordering constant $a$. Although it is possible to examine how the string spectrum changes as we vary these two parameters, the BRST quantization method we will analyze later fixes these two numbers uniquely. Without entering yet in the details of the BRST quantization of the string it is worth anticipating the two main outcomes of this process.

1. As for the particle and the more general example discussed in 2.3.1 we will have a BRST operator $Q_B$ which acts on an Hilbert space which is the tensor product of the Hilbert space we are considering now (the matter Hilbert space) and the ghost Hilbert space. Because of normal ordering subtleties the nilpotency of $Q_B$ will not be guaranteed, but we will have

$$Q_B^2 = 0 \quad \leftrightarrow \quad D = 26\,. \tag{3.105}$$

2. In this matter-ghost Hilbert space we can search for physical states by analyzing the kernel of $Q_B$ in a subspace of the form $|\Psi\rangle = |\psi\rangle^{\text{matter}} \otimes |\downarrow\rangle^{\text{ghost}}$, where $|\downarrow\rangle^{\text{ghost}}$ is a stringy analog of the corresponding ghost vacuum for the particle (2.83). We will then find

$$Q_B|\Psi\rangle = 0 \quad \Rightarrow \quad \begin{cases} L_n|\psi\rangle^{\text{matter}} = 0\,, & \forall n > 0\,, \\ (L_0 - 1)|\psi\rangle^{\text{matter}} = 0\,. \end{cases} \tag{3.106}$$

Which are the OCQ constraints we have been discussing above, but with the precise choice $a = 1$.

### 3.2.1.5 Physical, spurious and null states

Continuing with our OCQ framework, it is useful to introduce some definitions.

- A *physical* state $|\text{phys}\rangle$ is a state that satisfies the physical conditions

$$L_n|\text{phys}\rangle = 0\,, \quad \forall n > 0\,, \tag{3.107a}$$
$$(L_0 - a)|\text{phys}\rangle = 0\,. \tag{3.107b}$$

- A *spurious* state $|\chi\rangle$ is a state generated by negatively-moded Virasoros $L_{-n}$ acting an a generic state $|\chi_n\rangle$):

$$|\chi\rangle = \sum_{n>0} L_{-n}|\chi_n\rangle\,. \tag{3.108}$$

Such a state has zero inner product with any physical state:

$$\langle\chi|\text{phys}\rangle = 0\,. \tag{3.109}$$

- A *null* state $|\text{null}\rangle$ is a state that is both physical and spurious:

$$|\text{null}\rangle = \sum_{n>0} L_{-n} |\chi_n\rangle \,, \tag{3.110a}$$

$$L_n |\text{null}\rangle = 0 \,, \quad \forall n > 0 \,, \tag{3.110b}$$

$$(L_0 - a) |\text{null}\rangle = 0 \,. \tag{3.110c}$$

A null state has zero inner product with other null states and with physical states:

$$\langle \text{null} | \text{null}' \rangle = 0 \,, \qquad \langle \text{null} | \text{phys} \rangle = 0 \,. \tag{3.111}$$

This suggests to define an equivalence relation inside the space of physical states, stating the fact that the inner product between physical states does not change if we add null states to the physical states:

$$\boxed{|\text{phys}\rangle \sim |\text{phys}\rangle + |\text{null}\rangle \,.} \tag{3.112}$$

As we will see, the freedom of adding null states to physical states will be perceived in the target space as an (on-shell) *space-time gauge invariance*.

### 3.2.1.6 Physical spectrum of the closed string

Let us now define the level operator

$$\overset{(\sim)}{N} = \sum_{k>0} \overset{(\sim)}{\alpha}_{-k} \cdot \overset{(\sim)}{\alpha}_k \,. \tag{3.113}$$

This operator counts the level of oscillator excitation, by assigning to each oscillator $\alpha_{-n}$ a level $n$. For example we have

$$N \left(\alpha_{-1}^{\mu}\right)^2 \left(\alpha_{-2}^{\nu}\right)^3 |0, P\rangle = 8 \left(\alpha_{-1}^{\mu}\right)^2 \left(\alpha_{-2}^{\nu}\right)^3 |0, P\rangle \,, \tag{3.114a}$$

$$N \left(\alpha_{-3}^{\mu}\right) \left(\alpha_{-2}^{\nu}\right) |0, P\rangle = 5 \left(\alpha_{-3}^{\mu}\right) \left(\alpha_{-2}^{\nu}\right) |0, P\rangle \,. \tag{3.114b}$$

We can then write (see eq. (3.95))

$$\boxed{\overset{(\sim)}{L}_0 = \frac{\alpha' P^2}{4} + \overset{(\sim)}{N} \,.} \tag{3.115}$$

We will study the Hilbert space by decomposing it in eigenspaces of $(N, \widetilde{N})$ with fixed level. This is useful because these eigenspaces are finite dimensional (compared to the infinite dimensionality of the full Hilbert space).

Before we move forward with the analysis of the spectrum, it is useful to rewrite the $(L_0, \tilde{L}_0)$ constraints as

$$\left(L_0 + \widetilde{L}_0\right) |0, P\rangle = \left(\frac{\alpha' P^2}{2} + N + \widetilde{N}\right) |0, P\rangle$$

$$= \left(-\frac{\alpha' m^2}{2} + N + \widetilde{N}\right) |0, P\rangle = 2a |0, P\rangle \,, \tag{3.116a}$$

$$\left(L_0 - \widetilde{L}_0\right) |0, P\rangle = \left(N - \widetilde{N}\right) |0, P\rangle = 0 \,, \tag{3.116b}$$

where we assumed $a = \tilde{a}$ (remember that $a = \tilde{a} = 1$). If we have a generic state $|N, \widetilde{N}\rangle$ of level $(N, \widetilde{N})$, the constraint (3.116b) reads as the level matching condition:

$$N = \widetilde{N}.$$

(3.117)

It simply establishes that a physical state has to contain the same oscillator level on both sides. Therefore, instead of $(N, \widetilde{N})$, we will simply use $N$ to denote the level of the state of interest.

On the other hand, using the mass-shell condition $p^2 = -m^2$ the constraint (3.116a) reads

$$-\frac{\alpha' m^2}{2} + 2N = 2a,$$

(3.118)

and hence, using (3.117), it gives the mass-shell of the state $|N, N\rangle$:

$$\alpha' m^2 = 4(N - a).$$

(3.119)

We can now move on and analyze the first levels.

■ **Level $N = 0$**

We start from $N = 0$, i.e. by $|0, P\rangle$. We have that

$$\overset{(\sim)}{L}_n |0, P\rangle = 0, \quad \forall n > 0,$$

(3.120a)

$$\overset{(\sim)}{L}_0 |0, P\rangle = \left( \frac{\alpha' P^2}{4} + 0 \right) |0, P\rangle = a |0, P\rangle .$$

(3.120b)

The mass-shell condition ($p^2 = -m^2$) together with the second constraint (i.e. (3.120b)) gives

$$-\frac{\alpha' m^2}{4} = a \implies m^2 = -\frac{4a}{\alpha'}.$$

(3.121)

If $a > 0$, the $(0, 0)$ state has $m^2 < 0$, namely it is a tachyon. Since, at the end of the day $a = 1$, we definitely have a tachyon in the spectrum. This is not a good news because it is saying that we are quantizing the theory around an *unstable vacuum*. The question whether is there a new stable vacuum in the tachyon potential (just like the stable symmetry-breaking vacuum of the Higgs field of the Standard Model) is as of today an *open question* and a great mystery. Our (pedagogical) strategy in the sequel will be to *ignore* this tachyon problem and use this model to learn as much as we can in this simple bosonic setting. Later on we will perform a similar analysis for the (more complicated) *super-string* model where the tachyon will not be there and where genuine space-time fermions will arise in the spectrum. But for the time being we continue with the bosonic string.

■ **Level $N = 1$**

At the level $N = 1$ the only possible state is

$$\alpha_{-1}^{\mu} \tilde{\alpha}_{-1}^{\nu} |0, P\rangle ,$$

(3.122)

whose mass is $\alpha' m^2 = 4(1 - a)$. This state has two Lorentz indices of undefined symmetry, therefore it is a reducible representation of the Lorentz group. If we saturate them with a spacetime polarization (depending on the momentum), we get

$$G_{\mu \nu}(P) \alpha_{-1}^{\mu} \tilde{\alpha}_{-1}^{\nu} |0, P\rangle .$$

(3.123)

SciPost Phys. Lect. Notes 90 (2025)

The tensor $G_{\mu\nu}$ can be split into three Lorentz irreducible representations

$$G_{\mu\nu} = h_{\mu\nu} + B_{\mu\nu} + \Phi\eta_{\mu\nu}, \tag{3.124}$$

defined as follows:

$$\begin{cases} h_{\mu\nu} = G_{(\mu\nu)} \underbrace{-\frac{1}{D}\eta^{\mu\nu}G_\rho{}^\rho}_{\text{remove the trace}}, & \text{symmetric traceless,} \\ B_{\mu\nu} = G_{[\mu\nu]}, & \text{anti-symmetric,} \\ \Phi = \underbrace{\frac{1}{d}G_\rho{}^\rho}_{\text{trace}}, & \text{scalar.} \end{cases} \tag{3.125}$$

The most generic state can be written integrating over the momentum:

$$|\psi\rangle = \int d^d P \, G_{\mu\nu}(P)\alpha_{-1}^\mu \tilde{\alpha}_{-1}^\nu |0, P\rangle. \tag{3.126}$$

If we now consider the $n \neq 0$ Virasoro constraints (see eq. (3.103)), we have that only $L_1$ gives a non-trivial contribution, while $L_{n\geq 2}\alpha_{-1}^\mu \tilde{\alpha}_{-1}^\nu |0, P\rangle = 0$. We find indeed

$$\begin{aligned} L_1 G_{\mu\nu}\alpha_{-1}^\mu \tilde{\alpha}_{-1}^\nu |0, P\rangle &= 0 \\ &\propto G_{\mu\nu}(\alpha_0^\rho \alpha_1^\sigma)\eta_{\rho\sigma}\alpha_{-1}^\mu \tilde{\alpha}_{-1}^\nu |0, P\rangle \\ &= G_{\rho\nu}P^\rho \tilde{\alpha}_{-1}^\nu |0, P\rangle. \end{aligned} \tag{3.127}$$

Therefore $L_1$ and $\widetilde{L}_1$ impose (respectively)

$$\begin{cases} P_\mu G^{\mu\nu} = 0, \\ G^{\mu\nu}P_\nu = 0. \end{cases} \tag{3.128}$$

On the other hand, the mass-shell constraint (3.119) says that

$$\alpha' m^2 = 4(1-a), \tag{3.129}$$

which means that for $a < 1$ we have massive states, for $a = 1$ we have massless states and for $a > 1$ we have tachyons (that are now tensors, and not scalars, as in the $(0,0)$ case). Let's analyze the situation in detail.

- We may start considering the $a > 1$ case, where the states are tachyons. Since $P^2 > 0$, $P^\mu$ is a space-like vector and therefore there is a certain reference frame where it can be written as $P^\mu = (0, \vec{P})$. Moreover, the condition $P_\mu G^{\mu\nu} = 0$ implies that we can take an acceptable $G^{\mu\nu} = \delta^{\mu 0}\xi^\nu$. We can then consider the state

$$|\text{state}\rangle = \xi_\nu \alpha_{-1}^0 \tilde{\alpha}_{-1}^\nu |0, P\rangle, \tag{3.130}$$

which is physical if

$$\frac{\alpha' P^2}{4} = a - 1 > 0. \tag{3.131}$$

In addition, $G^{\mu\nu}P_\nu = 0$ also implies that $\xi = (\xi^0, \vec{\xi})$, with $\vec{P} \cdot \vec{\xi} = 0$. We have thus a physical state. Now we can compute its norm, finding

$$|| |\text{state}\rangle ||^2 = \eta^{00}(\xi \cdot \xi)\delta(P + P') < 0. \tag{3.132}$$

Therefore for $a > 1$ there exist negative-norm states, which are inconsistent with the basic axioms of quantum mechanics. Therefore we must have $a \leq 1$.

- Let us now consider the $a = 1$ case, where $m^2 = 0$ and therefore

$$P^2 = 0. \tag{3.133}$$

In this sector we have physical zero-norm states. If we indeed consider the polarization

$$G_*^{\mu\nu} = P^\mu \xi^\nu + v^\mu P^\nu, \tag{3.134}$$

with $P \cdot \xi = 0$ and $v \cdot P = 0$, the state is physical and we also have

$$\left|\left| G_*^{\mu\nu} \alpha_{-1}^\mu \tilde{\alpha}_{-1}^\nu |0, P\rangle \right|\right|^2 \sim P^2 = 0. \tag{3.135}$$

This is a physical state having vanishing norm, but it is spurious too because

$$P^\mu \xi^\nu \alpha_{-1}^\mu \tilde{\alpha}_{-1}^\nu |0, P\rangle \propto L_{-1}(\xi \cdot \tilde{\alpha}_{-1}) |0, P\rangle, \tag{3.136a}$$

$$v^\mu P^\nu \alpha_{-1}^\mu \tilde{\alpha}_{-1}^\nu |0, P\rangle \propto (v \cdot \alpha_{-1}) \tilde{L}_{-1} |0, P\rangle. \tag{3.136b}$$

We hence found a null state, giving an equivalence relation on the physical states' space:

$$|\text{phys}\rangle \sim |\text{phys}\rangle + |\text{null}\rangle, \tag{3.137}$$

$$G_{\mu\nu} \sim G_{\mu\nu} + P^\mu \xi^\nu + v^\mu P^\nu \tag{3.138}$$

$$\left(\text{with } P \cdot \xi = 0, \ v \cdot P = 0, \ P^2 = 0\right).$$

We have that (3.138) is a gauge transformation of the polarization tensor $G_{\mu\nu}$. If we then split it into its three components (see (3.125)), we can write

$$h_{\mu\nu} \sim h_{\mu\nu} + P_{(\mu} \theta_{\nu)}, \tag{3.139}$$

$$B_{\mu\nu} \sim B_{\mu\nu} + P_{[\mu} \Lambda_{\nu]}, \tag{3.140}$$

$$\Phi \sim \Phi, \tag{3.141}$$

where we defined

$$\theta_\mu = \frac{\xi_\mu + v_\mu}{2}, \qquad \Lambda_\mu = \frac{\xi_\mu - v_\mu}{2}. \tag{3.142}$$

We then find the following states.

- The *graviton* $h_{\mu\nu}$ with its gauge transformation (see eq. 3.139) and with its physical conditions

$$\begin{cases} P^2 h_{\mu\nu} = 0, \\ P_\mu h^{\mu\nu} = 0, \\ h_\mu{}^\mu = 0, \end{cases} \tag{3.143}$$

which imply (and in fact are equivalent to) the linearized Einstein equations in momentum space:[13]

$$P^2 h_{\mu\nu} - P^\rho P_{(\mu} h_{\nu)\rho} - P_\mu P_\nu h_\rho^\rho = 0. \tag{3.144}$$

---

[13]Similarly, the photon's EOMs are

$$\Box A_\mu - \partial_\mu (\partial \cdot A) = 0 \ \leftrightarrow \ P^2 A_\mu - P_\mu (P \cdot A) = 0.$$

Since $A_\mu$ and $P_\mu$ are in general two vectors pointing in different directions (an $A_\mu$ proportional to $P_\mu$ would be gauge trivial), the above equation is equivalent to

$$\begin{cases} P^2 = 0, \\ P \cdot A = 0. \end{cases}$$

– The *Kalb-Ramond field* $B_{\mu\nu}$, which is a 2-form

$$B = B_{\mu\nu}dx^\mu \wedge dx^\nu. \tag{3.145}$$

It comes with its own gauge transformation (see eq. 3.140) and its physical conditions are

$$\begin{cases} P^2 B_{\mu\nu} = 0, \\ P_\mu B^{\mu\nu} = 0, \end{cases} \tag{3.146}$$

which are equivalent to the EOMs of a 2-form which are briefly described as follows. We first define the field-strength

$$\begin{aligned} H = dB &= \partial_{[\mu}B_{\nu\rho]}dx^\mu \wedge dx^\nu \wedge dx^\rho \\ &= H_{\mu\nu\rho}dx^\mu \wedge dx^\nu \wedge dx^\rho. \end{aligned} \tag{3.147}$$

In complete analogy with the 1-form Maxwell field $A_\mu$ and its field-strenght $F_{\mu\nu} = \partial_{[\mu}A_{\nu]}$, the action functional for the 2-form field $B$ is given by[14]

$$S \sim \int d^D x\, H_{\mu\nu\rho}H^{\mu\nu\rho} \sim \int H \wedge *H, \tag{3.148}$$

$$\begin{cases} *d*H = 0 & \to\ \Box B_{\mu\nu} + \partial^\rho \partial_{[\mu}B_{\nu]\rho} = 0, \\ dH = 0 & \text{(Bianchi identities, no condition on } B). \end{cases} \tag{3.149}$$

– The *dilaton* $\Phi$, which is a gauge invariant (see eq. 3.141) scalar field which should obey the physical state condition

$$P^2 \Phi(P) = 0. \tag{3.150}$$

- Let us finally consider the $a < 1$ case. Here we have massive representations of the Lorentz group and therefore we do not find negative-norm physical states. In principle this would be acceptable. However, as already anticipated, the BRST quantization fixes $a = 1$ unambiguosly. Since, as we will see, we eventually need the BRST model to describe strings interactions, we will not consider this possibility further.

We hence found that the closed string spectrum at the $(1, 1)$ level is physically acceptable for $a \le 1$. At the critical value $a = 1$ there are zero-norm states, corresponding to the gauge symmetries. From a purely OCQ perspective $a = \tilde{a} = 1$ is required to have a properly defined graviton (plus dilaton and Kalb-Ramond) with their own gauge transformations, associated to null states. While, at this stage, this is still not sufficient to fix $a = 1$ by consistency it is however an indication that the correct choice $a = 1$ is associated with the appearance of null states in the spectrum.

■ **The Kalb-Ramond field and the fundamental string charge**

The fact that we have found a massless two-form in the closed string spectrum is deeply related to the one-dimensional nature of the string. We all know that when we have a charged particle of charge $q$, there is a minimal coupling between the particle world-line and the gauge field $A_\mu$ which feels the charge $q$:

$$q \int_{WL} A = q \int d\tau \dot{X}^\mu(\tau) A_\mu(X). \tag{3.151}$$

---

[14]Recall that in flat $D$-dimensional Minkowski space, the *Hodge dual* of a $p$-form $A_p = A_{\mu_1\cdots\mu_p}dx^{\mu_1} \wedge \cdots \wedge dx^{\mu_p}$, is the $(D-p)$-form $*A_p = A^{\mu_1\cdots\mu_p}\epsilon_{\mu_1\cdots\mu_d}dx^{\mu_{p+1}} \wedge \cdots \wedge dx^{\mu_D}$. This operation needs a spacetime metric, since the indices of $A$ are raised.

At the same time the charge $q$ is detectable by computing the flux of the electric field $E \sim *F$ over a sphere surrounding the particle:

$$q \sim \int_{S^{D-2}} *F^{(2)}, \tag{3.152}$$

where $*$ is the Hodge dual (which, in 4 dimensions, exchanges $\vec{E}$ with $\vec{B}$). This is Gauss theorem, appropriately generalized for a particle in $D$ space-time dimensions.

In the same way we can consider the Kalb-Ramond 2-form to minimally couple to the string itself via

$$-\frac{1}{4\pi\alpha'} \int_{WS} B = -\frac{1}{4\pi\alpha'} \int_{WS} d^2\sigma \, \epsilon^{\alpha\beta} \partial_\alpha X^\mu \partial_\beta X^\nu B_\mu(X), \tag{3.153}$$

and the fundamental string charge is then given by the flux of $H = dB$ through a spatial $S^{d-3}$-sphere encircling the string

$$\text{String charge} \sim \int_{S^{d-3}} *H^{(3)}. \tag{3.154}$$

■ **Higher levels**

Having fixed $a = 1$, at higher levels we find massive representations of the Lorentz group with, in general, spin higher than 2. The structure of these massive modes is quite complicated, especially for the closed string and it would not be so illuminating to discuss them in detail. We will analyze some of this massive states when we will study the easier open string spectrum. Here we would like to take the opportunity to learn that, just as the search for null states at level 1 selected $a = 1$, the search for null states at level 2 will instead select the critical dimension $D = 26$.

In order to see this, let us consider

$$|\chi_2\rangle = \underbrace{\left(2L_{-2} + 3L_{-1}^2\right)}_{\substack{\text{left} \\ \text{excitation}}} \underbrace{\left(\tilde{\theta}_{-2}\right)}_{\substack{\text{right} \\ \text{excitation}}} |0, P\rangle, \tag{3.155}$$

where $\tilde{\theta}_{-2}$ is a generic operator made up of right oscillators $\tilde{\alpha}$ (adding up to $\tilde{N} = 2$).

Setting $a = \tilde{a} = 1$, we will assume that $\tilde{\theta}_{-2}$ has been chosen so that the $\tilde{L}_n$ Virasoro constraints are already satisfied:

$$(\tilde{L}_0 - 1)\tilde{\theta}_{-2}|0, P\rangle = 0, \tag{3.156a}$$

$$\tilde{L}_1 \tilde{\theta}_{-2}|0, P\rangle = 0, \tag{3.156b}$$

$$\tilde{L}_2 \tilde{\theta}_{-2}|0, P\rangle = 0. \tag{3.156c}$$

Notice that, thanks to the Virasoro algebra, it is enough to impose $\tilde{L}_1 = \tilde{L}_2 = 0$ to guarantee that $\tilde{L}_{n>0} = 0$.

> **Exercise 3.2.4**
>
> Prove this.

Now we would like to check whether this spurious state is also physical, i.e. it is null. Explicit calculation shows that

$$(L_0 - 1)|\chi_2\rangle = \left(\frac{\alpha' P^2}{4} + 1\right)|\chi_2\rangle = 0 \rightarrow m^2 = \frac{4}{\alpha'}, \tag{3.157a}$$

$$L_1|\chi_2\rangle = 0, \tag{3.157b}$$

$$L_2|\chi_2\rangle \propto (D - 26)\tilde{\theta}_{-2}|0, P\rangle. \tag{3.157c}$$

**Exercise 3.2.5**

Show this.

Moreover, one can show that

$$|| |\chi\rangle ||^2 \propto (26 - D) . \tag{3.158}$$

Therefore the norm is zero for $D = 26$, precisely when the state is null, as expected.

It can be shown that when $a = 1$ and $D = 26$ there is a proliferation of similar null states which, as previously argued, define equivalence relations between physical states. This is the so-called critical string theory and these null states are the hallmark of the BRST cohomology of the string which will be indeed consistently realized only for $a = 1$ and $D = 26$.

To better appreciate the structure of level-two null states you can amuse yourself with the following exercise.

**Exercise 3.2.6**

In the left moving sector, consider the level two state

$$|\Psi\rangle = \left( \alpha_{-1} \cdot \alpha_{-1} + c_2 \alpha_0 \cdot \alpha_{-2} + 2c_3 (\alpha_0 \cdot \alpha_{-1})^2 \right) |0, P\rangle . \tag{3.159}$$

Determine $c_2$ and $c_3$ in such a way that the state is physical (for arbitrary value of the normal ordering constant $a$). For the physical values of $c_2$ and $c_3$ show that the norm of the state is given by

$$\langle \Psi | \Psi \rangle = \frac{2(D-1)[2(a-2)(8a-21)+(2a-3)D]}{(9-4a)^2} \langle 0, P | 0, P \rangle . \tag{3.160}$$

Therefore (assuming $D > 1$) we have that

$$\langle \Psi | \Psi \rangle \geq 0 \quad \rightarrow \quad D \leq \frac{2(2-a)(21-8a)}{3-2a} . \tag{3.161}$$

### 3.2.2 OCQ for the open string

#### 3.2.2.1 Classical solution and quantization

Let us now focus on the open string case. As we have seen, the EOMs are (3.51)

$$\begin{cases} \partial_+\partial_- X^\mu(\sigma^+, \sigma^-) &= 0 \qquad \text{(WS bulk eq.)}, \\ \delta X_\mu \partial_\sigma X^\mu \Big|_{\sigma=0,\pi} &= 0 \qquad \text{(WS boundary eq.)}. \end{cases} \tag{3.162}$$

The open string's endpoints (corresponding to $\sigma = 0$ and $\sigma = \pi$) are characterized by their boundary conditions (BC) which, in this case, can be Neumann or Dirichlet.[15]

- The Neumann (N) boundary condition is

$$\partial_\sigma X^\mu \Big|_{\text{endpoint}} = 0 , \tag{3.163}$$

which preserves Poincaré symmetry. It correspond to the statement that there is no momentum transfer along a certain direction $\mu$ in spacetime.

---

[15]In a more generic string background there can be many different, equally consistent, boundary conditions. Neumann and Dirichlet are the only possible boundary conditions for a single scalar field $X^i$ in a non-compact space.

- The Dirichlet (D) boundary condition is

$$\delta X_\mu\Big|_{\text{endpoint}} = 0\,, \tag{3.164}$$

  which breaks translational symmetry, since it corresponds to the statement that the endpoint is fixed at a certain point along the direction $\mu$ in spacetime.

**N-N string**

Let us first consider an open string having Neumann boundary conditions on both its endpoints along a certain direction $\mu$. We may then write the solution (in the Neumann direction $\mu$) as we did for the closed string, namely by splitting it into its left and right-moving parts, using lightcone coordinates:

$$X^\mu(\sigma^+, \sigma^-) = X_L^\mu(\sigma^+) + X_R^\mu(\sigma^-)\,. \tag{3.165}$$

We then define the auxiliary quantity

$$\hat{X}^\mu(\tau, \sigma) = \begin{cases} X^\mu(\tau, \sigma), & \text{if } \sigma \in (0, \pi), \\ X^\mu(\tau, 2\pi - \sigma), & \text{if } \sigma \in (\pi, 2\pi), \end{cases} \tag{3.166}$$

and naturally extend it by $2\pi$-periodicity:

$$\hat{X}^\mu(\tau, 2\pi + \sigma) = \hat{X}^\mu(\tau, \sigma)\,. \tag{3.167}$$

In addition, by construction, $\hat{X}$ also satisfies

$$\hat{X}(\tau, \sigma) = \hat{X}(\tau, -\sigma)\,. \tag{3.168}$$

Notice that $\hat{X}^\mu$ is analogous to the closed string solution. However the additional invariance $\sigma \to -\sigma$ implies that this time we have only a set of oscillators

$$\alpha_n^\mu = \tilde{\alpha}_n^\mu\,,$$

and the full solution then reads

$$X_{NN}^\mu(\sigma^+, \sigma^-) = X_0^\mu + \alpha' P^\mu \tau + i\sqrt{\frac{\alpha'}{2}} \sum_{n \neq 0} \frac{1}{n} \alpha_n^\mu \left( e^{-in\sigma^+} + e^{-in\sigma^-} \right)\,. \tag{3.169}$$

> **Exercise 3.2.7**
>
> Show that for such a solution the Neumann boundary condition $\partial_\sigma X^\mu\big|_{\sigma=0,\pi} = 0$ holds.

This time the Hilbert space will be acted by a single set of oscillators

$$\left[ \alpha_n^\mu, \alpha_m^\nu \right] = n\eta^{\mu\nu}\delta_{n+m,0}\,, \tag{3.170}$$

in addition with the center of mass Heisenberg algebra

$$[X_0^\mu, P^\nu] = i\eta^{\mu\nu}\,. \tag{3.171}$$

The generic state will hence be

$$\alpha_{-n_1}^{\mu_1} \dots \alpha_{-n_k}^{\mu_k} |0, P\rangle\,. \tag{3.172}$$

SciPost Phys. Lect. Notes 90 (2025)

In analogy with the closed string, the physical open string states can be defined by implementing the Virasoro constraints, using the quantum operators

$$L_{n\neq 0} = \frac{1}{2}\sum_{k\in\mathbb{Z}} \alpha_{n-k}\cdot\alpha_k\,, \tag{3.173a}$$

$$L_0 = \frac{1}{2}\sum_{k\in\mathbb{Z}} :\alpha_{-k}\cdot\alpha_k := \frac{1}{2}(\alpha_0)^2 + N = \alpha' P^2 + N\,, \tag{3.173b}$$

where in this case, for the NN open string, we define the zero-mode oscillator as

$$\alpha_0^\mu = \sqrt{2\alpha'}P^\mu \qquad \text{(open NN string)}. \tag{3.174}$$

**D-D string**

Let us now consider an open string having Dirichlet boundary conditions on both its endpoints along a certain direction $i$. Since $\delta X_i\big|_{\sigma=0,\pi} = 0$, we can fix

$$X^i(\sigma = 0) = x^i\,, \qquad X^i(\sigma = \pi) = x^i\,. \tag{3.175}$$

Therefore the solution (in the Dirichlet direction $i$) can be written as

$$X^i(\tau,\sigma) = x^i + Y^i(\tau,\sigma)\,, \tag{3.176}$$

where $Y^i(\tau,\sigma)$ is the oscillatory part and obeys

$$Y^i(\tau,\sigma)\Big|_{\sigma=0,\pi} = 0\,, \tag{3.177}$$

so that $\delta Y_i\big|_{\sigma=0,\pi} = 0$. We then define

$$\hat{Y}^i(\tau,\sigma) = \begin{cases} Y^i(\tau,\sigma)\,, & \text{if } \sigma\in(0,\pi)\,, \\ -Y^i(\tau,2\pi-\sigma)\,, & \text{if } \sigma\in(\pi,2\pi)\,, \end{cases} \tag{3.178}$$

and extend it by $2\pi$-periodicity. This time $\hat{Y}$ is odd under $\sigma\to-\sigma$:

$$\hat{Y}^i(\tau,-\sigma) = \hat{Y}^i(\tau,2\pi-\sigma) = -\hat{Y}^i(\tau,\sigma)\,. \tag{3.179}$$

Therefore the solution for the oscillatory part can be written as a closed string with $\alpha_n = -\tilde{\alpha}_n$:

$$Y^i(\sigma^+,\sigma^-) = i\sqrt{\frac{\alpha'}{2}}\sum_{n\neq 0}\frac{1}{n}\alpha_n^i\left(e^{-in\sigma^+} - e^{-in\sigma^-}\right)\,, \tag{3.180}$$

and the complete solution reads as

$$X_{DD}^i(\tau,\sigma) = x^i + i\sqrt{\frac{\alpha'}{2}}\sum_{n\neq 0}\frac{1}{n}\alpha_n^i\left(e^{-in\sigma^+} - e^{-in\sigma^-}\right)\,. \tag{3.181}$$

The term $\alpha' P^\mu\tau$ (containing the momentum of the center of mass) is absent, since $\tau$ would violate the anti-symmetric property of the solution $X^i(\tau,\sigma)$ under $\sigma\to-\sigma$. Therefore, when the open string's endpoints are fixed, the center of mass does not move.

**N-D string**

Let us finally consider an open string having Neumann boundary condition on one endpoint and Dirichlet boundary condition on the other one, along a certain direction $\mu$:

$$\partial_\sigma X^\mu \Big|_{\sigma=0} = 0, \qquad X^\mu(\tau, \sigma) \Big|_{\sigma=\pi} = x^\mu. \tag{3.182}$$

The solution can be written using half-integer moded oscillators as

$$X_{ND}^\mu(\tau, \sigma) = x^\mu + i\sqrt{\frac{\alpha'}{2}} \sum_{r \in \mathbb{Z}+\frac{1}{2}} \frac{1}{r} \alpha_r^\mu \left( e^{-ir\sigma^+} + e^{-ir\sigma^-} \right). \tag{3.183}$$

It has no momentum $P^\mu$ since one endpoint is fixed by the Dirichlet condition and hence the center of mass cannot have definite momentum.

Analogously, if we exchange the N and D conditions for $\sigma = 0$ and $\sigma = \pi$ we find

$$X_{DN}^\mu(\tau, \sigma) = X_0^\mu + i\sqrt{\frac{\alpha'}{2}} \sum_{r \in \mathbb{Z}+\frac{1}{2}} \frac{1}{r} \alpha_r^\mu \left( e^{-ir\sigma^+} - e^{-ir\sigma^-} \right). \tag{3.184}$$

Therefore the exchange N $\longleftrightarrow$ D translates into a change of sign of the anti-holomorphic sector (i.e. the right-moving part).

> **Exercise 3.2.8**
>
> Prove this last statement.

We can then define (for $n \in \mathbb{Z}$)

$$L_n = \frac{1}{2} \sum_{r \in \mathbb{Z}+\frac{1}{2}} :\alpha_{n-r} \cdot \alpha_r:, \tag{3.185}$$

together with the commutators

$$\left[ \alpha_r^\mu, \alpha_s^\nu \right] = r \eta^{\mu\nu} \delta_{r+s, 0}, \tag{3.186a}$$

$$[L_n, L_m] = (n-m)L_{n+m} + \frac{1}{12}n(n^2-1)\delta_{n+m, 0}. \tag{3.186b}$$

These operators will act on a vacuum $|0\rangle_{ND}$ as

$$\alpha_r |0\rangle_{ND} = 0, \quad \forall r \in \mathbb{Z} + \frac{1}{2}, \quad r \geq \frac{1}{2}. \tag{3.187}$$

Before we go on it is useful to analyze the stress-energy tensor and its relation with the boundary conditions of the open string:

$$T_{\pm\pm}(\sigma^\pm) = -\sum_{n \in \mathbb{Z}} L_n e^{-in\sigma^\pm}. \tag{3.188}$$

If we notice that

$$\partial_+ X_{NN}(\sigma^+) = \partial_- X_{NN}(\sigma^-)\Big|_{\sigma=0,\pi}, \tag{3.189a}$$

$$\partial_+ X_{DD}(\sigma^+) = -\partial_- X_{DD}(\sigma^-)\Big|_{\sigma=0,\pi}, \tag{3.189b}$$

we realize that for N-N and for D-D the stress energy tensor satisfies

$$T_{++}(\sigma^+) = T_{--}(\sigma^-)\Big|_{\sigma=0,\pi}. \tag{3.190}$$

This is an example of a *gluing condition*. Notice that, independently on the boundary conditions for $X$, $T$ has always the same boundary conditions. Similarly, for N-D open strings it easy to see that

$$\partial_+ X_{ND}(\sigma^+) = \partial_- X_{ND}(\sigma^-)\Big|_{\sigma=0}, \tag{3.191a}$$

$$\partial_+ X_{ND}(\sigma^+) = -\partial_- X_{ND}(\sigma^-)\Big|_{\sigma=\pi}. \tag{3.191b}$$

Nevertheless, $T$ has always the same boundary condition (3.190).

---

**Exercise 3.2.9**

Prove eqs. (3.191).

---

### 3.2.2.2 Mixed boundary conditions and $D(p)$-branes

Let us now consider an open string whose endpoints have:

- Neumann BC along $\mu = 0, 1, \ldots, p$.

- Dirichlet BC along $i = p + 1, \ldots, D - 1$ (where $D$ is the spacetime dimension).

These endpoints can therefore move only in the $p+1$ Neumann directions, since they are fixed in the $(D - p - 1)$ Dirichlet directions. This means that we have a sub-manifold $\mathbb{R}^{1,p}$ of the spacetime $\mathbb{R}^{1,d-1}$ where the endpoints can move around (see fig. 3.4). Such a sub-manifold is called a Dirichlet brane, more precisely a $D(p)$-*brane*. Notice in particular that if the open string has N-N endpoints in every direction, the $D(p)$-brane is identified with the whole spacetime.

The existence of a $D(p)$-brane breaks the invariance of the closed string theory under translations. As in a spontaneously symmetry breaking vacuum we then expect that Goldstone bosons arise, describing the displacement of the brane along the directions where the translational invariance is broken.

As the $D(p)$-brane moves in spacetime, it spans a $p + 1$-dimensional surface, called *worldvolume* (see fig. 3.5).

### 3.2.2.3 Single $D(p)$-brane spectrum

Let us consider a generic state made up of Neumann oscillators $\alpha_n^\mu$ and Dirichlet oscillators $\alpha_n^i$ acting on the vacuum $|0, P\rangle$, where the spacetime momentum $P^\mu$ lies on the $D(p)$-brane, since along Dirichlet directions we have no momentum (see fig. 3.5). The zero mode Virasoro operator is in this case

$$L_0 = \alpha' P^2 + N^{(NN)} + N^{(DD)} = \alpha' P^2 + N^{(\text{tot})}. \tag{3.192}$$

This turns into the mass formula

$$\alpha' m^2 = N^{(\text{tot})} - a. \tag{3.193}$$

We can now analyze the first energy levels of the spectrum.

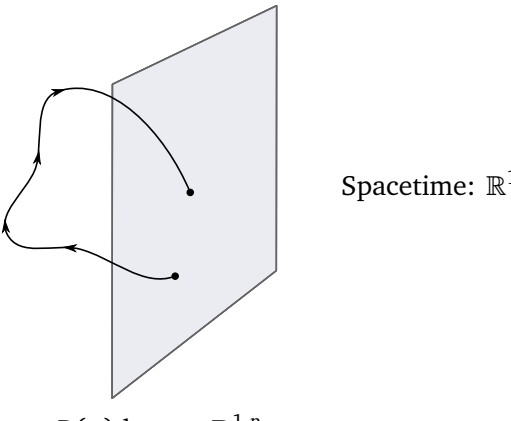

Figure 3.4: A $D(p)$-brane embedded in the target space $\mathbb{R}^{1,D-1}$.

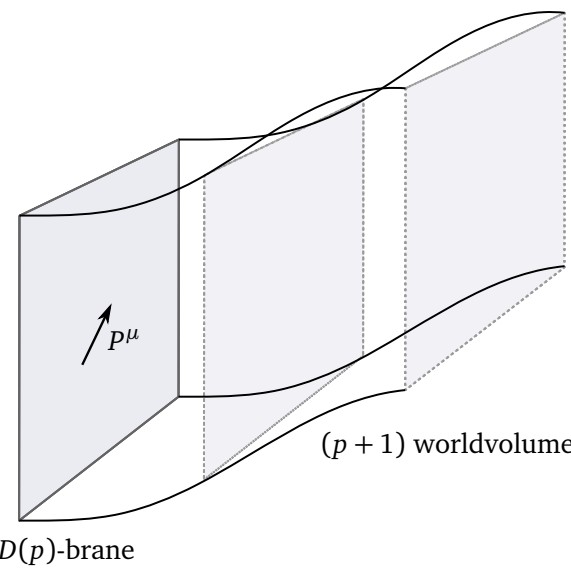

Figure 3.5: The trajectory of the $D(p)$-brane in the spacetime spans the worldvolume. The $p$-dimensional momentum $P^\mu$ lies on the $D(p)$-brane, since along Dirichlet directions we have no momentum.

■ **Level $N = 0$**

At the level $N = 0$ we have

$$t(P)|0,P\rangle\,, \tag{3.194}$$

where $t(P)$ is the spacetime polarization of the state. Its mass is given by

$$\alpha'm^2 = -a\,. \tag{3.195}$$

If we then take

$$a = 1\,, \tag{3.196}$$

this $N = 0$ state corresponds to a tachyon (whose spacetime polarization is $t(P)$) which is four times lighter than the closed tachyon (see eq. 3.121):

$$m^2 = -\frac{1}{\alpha'}\,. \tag{3.197}$$

■ **Level $N = 1$**

At the level $N = 1$ we have to take one oscillator $\alpha_{-1}$, that can excite the vacuum along a Neumann or a Dirichlet direction. If we saturate the free indices with a spacetime polarization, we have

$$\left(A_\mu(P)\alpha_{-1}^\mu + \phi_i(P)\alpha_{-1}^i\right)|0, P\rangle \,, \tag{3.198}$$

where

- $A_\mu(P)$ is a $p+1$-dimensional vector of $SO(1, p)$.

- $\phi_i(P)$ are $D - p - 1$ scalars, assembled in the vector representation of $SO(D - p - 1)$, which appears to be a global internal symmetry if we are sitting on the $D(p)$-brane.

Therefore the existence of the $D(p)$-brane breaks the spacetime Lorentz symmetry as follows:

$$SO(1, d-1) \longrightarrow \underbrace{SO(1, p)}_{\substack{\text{Lorentz on} \\ \text{the } D(p)\text{-brane}}} \times \underbrace{SO(d - p - 1)}_{\substack{\text{rotations in the} \\ \text{Dirichlet directions}}} \,. \tag{3.199}$$

If we then consider the mass, we have that $L_0 - a = 0 \implies \alpha'm^2 = 1 - a$, which for $a = 1$ translates into

$$m = 0 \,. \tag{3.200}$$

Therefore we found a massless gauge vector $A_\mu$ and $D - p - 1$ massless scalars $\phi_i$. These scalars are the Goldstone bosons in the directions where the translational symmetry has been broken by the $D(p)$-brane. By Goldstone theorem they should be massless and therefore the choice $a = 1$ we first pushed in (3.196) is in this case necessary.

These $\phi_i$ can be seen as the coordinates describing the motion of the $D(p)$-brane in the transverse directions, where it is embedded. Collectively, they can be seen as a map $\{\phi^i\} : \mathbb{R}^{1,p} \longrightarrow \mathbb{R}^{D-p-1}$ taking a point $x$ in the worldvolume of the $D(p)$-brane and saying where it lays in the transverse space, namely in the Dirichlet directions. When $\phi^i = 0$ (i.e. the scalar fields have zero vev), the $D(p)$-brane is not deformed.

If we now consider the Virasoro constraints $L_1 = 0$ and $L_0 = 1$, we find the EOMs for the massless gauge field (Maxwell's equations) and for the massless scalars (massless Klein-Gordon equation):

$$\begin{cases} P_\mu A^\mu(P) = 0 \,, \\ P^2 A^\mu(P) = 0 \,, \end{cases} \tag{3.201}$$

$$P^2 \phi^i(P) = 0 \,. \tag{3.202}$$

The equivalence relation (3.112) for the gauge field $A_\mu$ reads as the gauge invariance

$$A_\mu(P) \sim A_\mu(P) + \lambda(P)P_\mu \longrightarrow A_\mu(x) \sim A_\mu(x) + i\partial_\mu\lambda(x), \tag{3.203}$$

where $\lambda(P)P_\mu|0, P\rangle \sim L_{-1}|0, P\rangle$ is a null state, since $P^2 = 0$.

Again, at higher level, we find massive states, as described in the following exercise.

---

**Exercise 3.2.10**

Consider a $D(25)$-brane (Neumann conditions on all spacetime directions). Consider the most generic open string state at level $N = 2$

$$\left(\xi_\mu \alpha_{-2}^\mu + \theta_{\mu\nu} \alpha_{-1}^\mu \alpha_{-1}^\nu\right)|0, P\rangle \,, \tag{3.204}$$

and determine its physical conditions. Show that it is consistent to set $\xi_\mu = 0$ without loosing physical degrees of freedom and that in total we have a spin 2 massive particle (traceless symmetric rank two tensor).

---

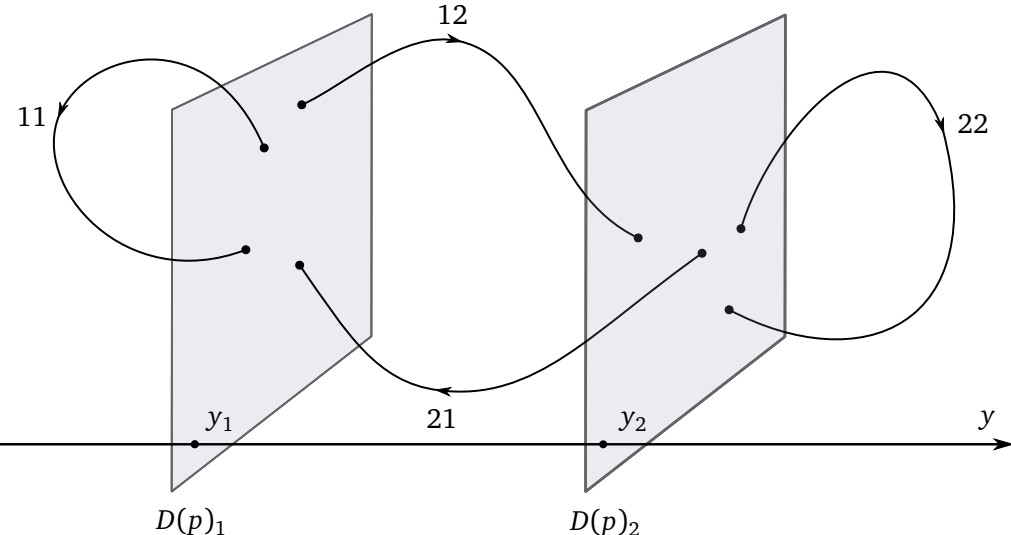

Figure 3.6: The $D(p) - D(p)$ system.

### 3.2.2.4 Multiple $D(p)$-branes spectrum

Let us now consider a D-brane system made up of two $D(p)$-branes ($D(p)_1$ and $D(p)_2$) parallel along a common Dirichlet direction $y$ (see fig. 3.6). They are both characterized by their Dirichlet moduli $y_i$, which are the precise values of the Dirichlet BC for the open strings attached to the $D(p)_i$-brane.

As already discussed in subsection 3.2.2.3, the spectrum (at the $N = 0, 1$ energy levels) on the two branes separately contains one tachyon (having $m^2 = -\frac{1}{\alpha'}$), one massless gauge field and $d - p - 1$ massless scalars. These are the states produced by strings that start and end on the same $D(p)$-brane. On the other hand, the strings that are stretched between the two $D(p)$-branes generate different states. They have the same BC along all the Dirichlet directions except for $y$, where the Dirichlet modulus changes at the two endpoints.

Let us first focus on the string that starts on the $D(p)_1$-brane and ends on the $D(p)_2$-brane. Recalling eq. (3.176), we can write its solution as $Y^{(12)}(\tau, \sigma)$. It is defined by

$$\partial_+ \partial_- Y^{(12)}(\tau, \sigma) = 0 \qquad \text{(wave equation)}, \qquad (3.205\text{a})$$

$$Y^{(12)}(\tau, \sigma = 0) = y_1 \qquad \text{(BC)}, \qquad (3.205\text{b})$$

$$Y^{(12)}(\tau, \sigma = \pi) = y_2 \qquad \text{(BC)}. \qquad (3.205\text{c})$$

The solution can be written as

$$Y^{(12)}(\sigma^+, \sigma^-) = \left(1 - \frac{\sigma}{\pi}\right) y_1 + \frac{\sigma}{\pi} y_2 + i\sqrt{\frac{\alpha'}{2}} \sum_{n \neq 0} \frac{1}{n} \alpha_n^{(12)} \left(e^{-in\sigma^+} - e^{-in\sigma^-}\right), \qquad (3.206)$$

and it is useful to write (with $\Delta y = y_2 - y_1$ being the distance between the two branes)

$$\left(1 - \frac{\sigma}{\pi}\right) y_1 + \frac{\sigma}{\pi} y_2 = y_1 + \frac{\Delta y}{2\pi}(\sigma^+ - \sigma^-). \qquad (3.207)$$

If we define the zero-mode oscillator as

$$\alpha_0^{(12)} = \frac{\Delta y}{2\pi}\sqrt{\frac{2}{\alpha'}}, \qquad (3.208)$$

we can evaluate

$$\partial_+ Y^{(12)}(\sigma^+,\sigma^-) = \frac{\Delta y}{2\pi} + \sqrt{\frac{\alpha'}{2}}\sum_{n\neq 0}\alpha_n^{(12)}e^{-in\sigma^+} \qquad = \sqrt{\frac{\alpha'}{2}}\sum_{n\in\mathbb{Z}}\alpha_n^{(12)}e^{-in\sigma^+}\,, \qquad (3.209a)$$

$$\partial_- Y^{(12)}(\sigma^+,\sigma^-) = -\frac{\Delta y}{2\pi} - \sqrt{\frac{\alpha'}{2}}\sum_{n\neq 0}\alpha_n^{(12)}e^{-in\sigma^-} \qquad = \sqrt{\frac{\alpha'}{2}}\sum_{n\in\mathbb{Z}}\alpha_n^{(12)}e^{-in\sigma^+}\,. \qquad (3.209b)$$

We can then build the stress-energy tensor in the $y$ direction (in the other directions the Dirichlet conditions are the same for both the endpoints of the stretched string) as follows:

$$T_{\pm\pm}^{(y)} = -\frac{1}{\alpha'}\partial_\pm Y \partial_\pm Y\,. \qquad (3.210)$$

We are now ready to study the spectrum of these stretched strings. The mass formula (setting $a = 1$) is

$$\alpha' m^2 = -1 + N^{(\text{tot})} + \frac{1}{2}\left(\frac{\Delta y}{2\pi}\right)^2\frac{2}{\alpha'} = -1 + N^{(\text{tot})} + \left(\frac{\Delta y}{2\pi\sqrt{\alpha'}}\right)^2\,. \qquad (3.211)$$

As $\Delta y$ increases, $m^2$ grows. Therefore the energy of the stretched string increases with the separation of the two $D$-branes.

■ **Level $N = 0$**
At the level $N = 0$ we have a scalar particle

$$t^{(12)}(P)|0,P\rangle\,, \qquad (3.212)$$

whose mass is given by

$$\alpha' m^2 = -1 + \left(\frac{\Delta y}{2\pi\sqrt{\alpha'}}\right)^2\,. \qquad (3.213)$$

Therefore, if the two $D(p)$-branes are at a distance $\Delta y$ such that

$$-1 + \left(\frac{\Delta y}{2\pi\sqrt{\alpha'}}\right)^2 \geq 0\,, \qquad (3.214)$$

the tachyon is not a tachyon anymore.

■ **Level $N = 1$**
At the level $N = 1$ we have

$$\left(W_\mu^{(12)}(P)\alpha_{-1}^\mu + \phi_i^{(12)}\alpha_{-1}^i\right)|0,P\rangle\,, \qquad (3.215)$$

whose mass is given by

$$\alpha' m^2 = \left(\frac{\Delta y}{2\pi\sqrt{\alpha'}}\right)^2\,. \qquad (3.216)$$

Therefore we have a massive vector $W_\mu^{(12)}$ and $D - p - 1$ massive scalars $\phi_i^{(12)}$. If we now consider the Virasoro constraints $L_1 = 0$ and $L_0 = 1$, we find the EOMs for the massive vector field (Proca's equations) and for the massive scalars (massive Klein-Gordon equations):

$$\begin{cases} P^\mu W_\mu^{(12)}(P) = 0\,, \\ \left(\alpha' P^2 + \left(\frac{\Delta y}{2\pi\sqrt{\alpha'}}\right)^2\right)W_\mu^{(12)} = 0\,, \end{cases} \qquad (3.217)$$

$$\left(\alpha' P^2 + \left(\frac{\Delta y}{2\pi\sqrt{\alpha'}}\right)^2\right)\phi_i^{(12)}(P) = 0\,. \qquad (3.218)$$

The $L_1$ constraint is subtle for $i = y$ (the direction along which we have displaced the $D(p)$-branes). It is not difficult to see that

$$L_1 \phi_y^{(12)}(P)\alpha_{-1}^y |0, P\rangle = \phi_y^{(12)}(\alpha_0^y \alpha_1^y)\alpha_{-1}^y |0, P\rangle \sim (\Delta y)\, \phi_y^{(12)}(P) |0, P\rangle \neq 0 \,. \tag{3.219}$$

Therefore if $\Delta y \neq 0$ the field $\phi_y^{(12)}(P)$ is not physical and thus disappears from the spectrum.

Everything we said so far for the (12) string (stretched between the $D(p)_1$ and the $D(p)_2$-brane) holds in the same way for the (21) string (stretched between the $D(p)_2$ and the $D(p)_1$-brane).

The generic field in such a $D(p)-D(p)$ system can be represented as a $2 \times 2$ matrix: on the diagonal we insert the strings having their endpoints both on the same $D(p)$-brane (namely the (11) and the (22) strings), while off the diagonal we locate the strings stretched between the two branes (namely the (12) and the (21) strings). The scalar fields' matrix is (for $\Delta y \neq 0$)

$$\begin{pmatrix} \phi_{i\neq y}^{(11)} & \phi_{i\neq y}^{(12)} \\ \phi_{i\neq y}^{(21)} & \phi_{i\neq y}^{(22)} \end{pmatrix}, \qquad \begin{pmatrix} \phi_y^{(11)} & 0 \\ 0 & \phi_y^{(22)} \end{pmatrix}, \qquad (\Delta y \neq 0). \tag{3.220}$$

Notice that there are not the $\phi_y^{(12)}$ and $\phi_y^{(21)}$ fluctuations as they cannot be physical when $\Delta y \neq 0$. The vector fields' matrix is

$$\begin{pmatrix} A_\mu^{(11)} & W_\mu^{(12)} \\ W_\mu^{(21)} & A_\mu^{(22)} \end{pmatrix}. \tag{3.221}$$

These matrices are called *Chan-Paton matrices*. Focusing on the latter, we can recall eq. (3.216) and notice that, when the two $D(p)$-branes are coincident (namely $\Delta y = 0$), all the vector fields are massless. This means that

$$\underbrace{U(1) \times U(1)}_{\substack{A_\mu \text{ massless} \\ W_\mu \text{ massive}}} \quad \xrightarrow{\Delta y = 0} \quad \underbrace{U(2)}_{\substack{A_\mu \text{ massless} \\ W_\mu \text{ massless}}} \quad . \tag{3.222}$$

Notice that in the $\Delta y \to 0$ limit the number of degrees of freedom is conserved: at finite separation we have two massive vector fields ($d-1$ degrees of freedom) which at vanishing separation become two massless vector fields ($d-2$ degrees of freedom). However in the same limit the two scalars $\phi_y^{12}$ and $\phi_y^{21}$ become physical as the $L_1$ constraint is now identically satisfied. Therefore a Higgs-like mechanism takes place by separating the two $D$-branes: the gauge symmetry $U(2)$ is broken to $U(1) \times U(1)$ and two Goldstone modes ($\phi_y^{12}$ and $\phi_y^{21}$) are 'eaten' by the broken gauge fields that acquire mass. This is all very nice and consistent!

If we have $M$ coincident $D(p)$-branes, then we get $M^2$ massless gauge bosons giving rise to a non-Abelian $U(M)$ gauge theory. This is not fully justified from the spectrum, but we will verify later that these massless vectors will indeed interact precisely as in a non-Abelian Yang-Mills theory.

## 3.3 BRST quantization of the string

It is now time to introduce the BRST formalism. We will follow the steps as in section 2.2 and we start by defining the Polyakov path integral as

$$Z = \frac{1}{\text{Vol}\mathcal{G}} \int \mathcal{D}X^\mu \mathcal{D}h_{\alpha\beta}\, e^{iS[X,h]} \,, \tag{3.223}$$

where $S[h,X]$ is the already introduced Polyakov action

$$S[h,X] = -\frac{1}{4\pi\alpha'} \int_{\text{WS}} d^2\sigma \sqrt{-h}\, h^{\alpha\beta}\, \partial_\alpha X^\mu \partial_\beta X_\mu\,. \tag{3.224}$$

To simplify out the volume of the gauge group we must fix the gauge symmetries, which are generated by the following two infinitesimal transformations:

1. Diffeomorphisms: $\delta_D X^\mu = \xi^\rho \partial_\rho X^\mu,\quad \delta_D h_{\alpha\beta} = \nabla_\alpha \xi_\beta + \nabla_\beta \xi_\alpha\,.$

2. Weyl transformations: $\delta_{\text{W}} h_{\alpha\beta} = \omega h_{\alpha\beta}$,

where we have defined the infinitesimal group parameters $\xi^\rho(\sigma,\tau)$, $\omega(\sigma,\tau)$, and the covariant derivative

$$\nabla_\alpha \xi_\beta = \partial_\alpha - \Gamma^\gamma_{\alpha\beta} \xi_\gamma\,, \tag{3.225}$$

with the Christoffel symbols

$$\Gamma^\gamma_{\alpha\beta} = \frac{1}{2} h^{\gamma\sigma} \left( \partial_\alpha h_{\sigma\beta} + \partial_\beta h_{\sigma\alpha} - \partial_\sigma h_{\alpha\beta} \right)\,. \tag{3.226}$$

The idea to fix these gauge redundancies is to fix the metric to a fiducial value. To do this we will assume that any metric $h$ can be obtained by making a gauge transformation to the fiducial metric $\hat{h}$. I.e. we will assume that

$$h = \hat{h}^g\,,$$

for some gauge transformation $g$. In particular the fiducial value $h = \hat{h}$ corresponds to $g = 1$. This essentially means that we consider *all* degrees of freedom of the metric to be pure gauge. To be precise, this is true if the worldsheet has the topology of a sphere (or a disk). But on a generic surface this is true only locally and indeed we will see that there are in general *metric moduli* which cannot be changed by diffeomorphisms. When this happens the functional integration over the metric will not completely cancel against the volume of the gauge group but it will leave out a *finite dimensional* integral on the metric moduli. While this is a major ingredient for the construction of strings' amplitudes, it is not going to affect the construction of the BRST charge and the spectrum so we will postpone this (however important) subtlety for later on. That said, our gauge fixing condition will be represented by the functional condition

$$F(h) = h - \hat{h} = 0 \;\rightarrow\; h = \hat{h}\,. \tag{3.227}$$

We can now write the identity

$$1 = \int \mathcal{D}h\, \delta(h - \hat{h}) = \int \mathcal{D}\hat{h}^g\, \delta(\hat{h}^g - \hat{h}) = \int \mathcal{D}g\, \delta(\hat{h}^g - \hat{h})\det \frac{\delta \hat{h}^g}{\delta g}\Big|_{g=1}\,, \tag{3.228}$$

where $g$ represents the gauge group element chosen to parametrize the gauge orbit of $\hat{h}^g$ so that $\hat{h}^1 = \hat{h}$. Using this resolution of the identity in 3.223 we can factorize the gauge volume

and obtain the gauge fixed Polyakov path integral

$$
\begin{aligned}
Z &= \frac{1}{\text{Vol}\mathcal{G}} \int \mathcal{D}X \mathcal{D}h \mathcal{D}g \; \delta(\hat{h}^g - \hat{h}) \det\frac{\delta\hat{h}^g}{\delta g}\bigg|_{g=1} e^{iS[X,h]} \\
&= \frac{1}{\text{Vol}\mathcal{G}} \int \mathcal{D}X \mathcal{D}\hat{h}^g \mathcal{D}g \; \delta(\hat{h}^g - \hat{h}) \det\frac{\delta\hat{h}^g}{\delta g}\bigg|_{g=1} e^{iS[X,\hat{h}^g]} \\
&= \frac{1}{\text{Vol}\mathcal{G}} \int \mathcal{D}X^g \mathcal{D}\hat{h}^g \mathcal{D}g \; \delta(\hat{h}^g - \hat{h}) \det\frac{\delta\hat{h}^g}{\delta g}\bigg|_{g=1} e^{iS[X^g,\hat{h}^g]} \\
&= \frac{1}{\text{Vol}\mathcal{G}} \int \mathcal{D}X^g \mathcal{D}\hat{h}^g \mathcal{D}g \; \delta(\hat{h}^g - \hat{h}) \det\frac{\delta\hat{h}^g}{\delta g}\bigg|_{g=1} e^{iS[X,\hat{h}]} \\
&= \frac{1}{\text{Vol}\mathcal{G}} \int \mathcal{D}X \mathcal{D}g \; \det\frac{\delta\hat{h}^g}{\delta g}\bigg|_{g=1} e^{iS[X,\hat{h}]} \\
&= \int \mathcal{D}X \; \det\frac{\delta\hat{h}^g}{\delta g}\bigg|_{g=1} e^{iS[X,\hat{h}]},
\end{aligned}
\tag{3.229}
$$

where in the second line we have written $h$ as $\hat{h}^g$, in the third line we have changed the dummy integration variable $X$ to $X^g$, in the fourth line we have used the gauge invariance of the action, in the fifth line we have used the gauge invariance of the measure $\mathcal{D}X$ and we have eliminated the delta function. Finally in the last line we have simplified the volume of the gauge group with $\int \mathcal{D}g$ thanks to the gauge invariance of the determinant.

By formally rewriting the determinant as a Berezin integral (Fadeev-Popov trick) we obtain the FP action

$$
\det\frac{\delta h^g}{\delta g}\bigg|_{g=1} = \int \mathcal{D}b_{\alpha\beta} \mathcal{D}\lambda \mathcal{D}c^\rho \exp\left[\frac{1}{4\pi} \int d^2\sigma \sqrt{-\hat{h}} \, b_{\alpha\beta} \frac{\delta h^{\alpha\beta}}{\delta g^x}\bigg|_{g=1} c^x\right],
\tag{3.230}
$$

where we have defined $c^x = (\lambda, c^\rho)$ with $\lambda$ and $c^\rho$ respectively defined as the ghost fields related to the Weyl and the diffeos generators. It is usual to explicitly write the variation of the metric as

$$
\delta_{\text{tot}}h_{\alpha\beta} = \delta_D h_{\alpha\beta} + \delta_W h_{\alpha\beta} = \omega h_{\alpha\beta} + \nabla_\alpha \xi_\beta + \nabla_\beta \xi_\alpha = \Omega h_{\alpha\beta} + 2(P_1\xi)_{\alpha\beta},
\tag{3.231}
$$

where we have defined

$$
\Omega \equiv \omega + \nabla\cdot\xi,
\tag{3.232}
$$

and

$$
(P_1\xi)_{\alpha\beta} \equiv \frac{1}{2}\left(\nabla_\alpha \xi_\beta + \nabla_\beta \xi_\alpha - h_{\alpha\beta}\nabla\cdot\xi\right),
\tag{3.233}
$$

which is a map from vector fields (one index up) to traceless quadratic differentials (two lower symmetric indices with vanishing trace). With these definitions we can easily write down the ghost lagrangian by substituting the gauge parameters $(\omega, \xi^\alpha)$ with the corresponding Weyl and Diff ghosts $(\lambda, c^\alpha)$ in $\delta h^{\alpha\beta}$ and then contracting with the antighost $b_{\alpha\beta}$:

$$
\int d^2\sigma \sqrt{-\hat{h}} \, b_{\alpha\beta} \frac{\delta h^{\alpha\beta}}{\delta g^x}\bigg|_{g=1} c^x = \int d^2\sigma \sqrt{-\hat{h}} \, b_{\alpha\beta} \left[\hat{h}^{\alpha\beta}\left(\lambda + \frac{1}{2}\nabla\cdot c\right) + (2P_1c)^{\alpha\beta}\right].
\tag{3.234}
$$

Using this result we can integrate out the ghost $\lambda$ in the path integral which imposes that $b_{\alpha\beta}\hat{h}^{\alpha\beta} = 0$:

$$
Z = \int \mathcal{D}X^\mu \mathcal{D}b_{\alpha\beta} \mathcal{D}c^\rho \, e^{i\left(S[X,\hat{h}] - \frac{i}{4\pi}\int d^2\sigma \sqrt{-h} \, b_{\alpha\beta}(2P_1c)^{\alpha\beta}\right)},
\tag{3.235}
$$

where $b_{\alpha\beta}$ is now traceless. With this last condition we obtain the complete gauge fixed path integral:

$$Z_{Pol}[X^{\mu}, b_{\alpha\beta}, c^{\rho}] = \int \mathcal{D}X^{\mu} \mathcal{D}b_{\alpha\beta} \mathcal{D}c^{\rho} \; e^{i\left(S[X,\hat{h}] - \frac{i}{4\pi} \int d^2\sigma \sqrt{-\hat{h}} b_{\alpha\beta}(\nabla^{\alpha} c^{\beta} + \nabla^{\beta} c^{\alpha})\right)}. \tag{3.236}$$

We have found the explicit form of the ghost action to be

$$S_{\text{GH}} = -\frac{i}{4\pi} \int d^2\sigma \sqrt{-\hat{h}} b_{\alpha\beta}(\nabla^{\alpha} c^{\beta} + \nabla^{\beta} c^{\alpha}) = -\frac{i}{2\pi} \int D^2\sigma \sqrt{-\hat{h}} b_{\alpha\beta} \nabla^{\alpha} c^{\beta}, \tag{3.237}$$

where in the second step we have considered that $b_{\alpha\beta}$ is a symmetric tensor, since it only appears contracted with $\delta h^{\alpha\beta} = \delta h^{\beta\alpha}$.

It is interesting to notice that the ghost sector of the metric is still Weyl invariant. If we consider the transformation $h \to e^{-\omega} h$ we obtain

$$iS'_{\text{GH}} = \frac{1}{2\pi} \int d^2\sigma \sqrt{-h} e^{-\omega} e^{\omega} h^{\alpha\beta} b_{\beta\delta} \nabla'_{\alpha} c^{\delta} \tag{3.238}$$

$$= \frac{1}{2\pi} \int d^2\sigma \sqrt{-h} h^{\alpha\beta} b_{\beta\delta} \left(\partial_{\alpha} c^{\delta} + \Gamma^{\delta}_{\alpha\sigma}{}' c^{\sigma}\right)$$

$$= -S_{\text{GH}} - \frac{1}{4\pi} \int d^2\sigma \sqrt{-h} h^{\alpha\beta} b_{\beta\delta} \left(h^{\delta}_{\alpha} \partial_{\sigma} \omega + h^{\delta}_{\sigma} \partial_{\alpha} \omega - h_{\alpha\sigma} \partial^{\delta} \omega\right) c^{\sigma}$$

$$= iS_{\text{GH}} - \underbrace{\frac{1}{4\pi} \int d^2\sigma \sqrt{-h} b^{\alpha}{}_{\alpha} \partial_{\sigma} \omega c^{\sigma}}_{=0 \text{ because } b^{\alpha}_{\alpha}=0} - \underbrace{\frac{1}{4\pi} \int d^2\sigma \sqrt{-h} \left(b^{\alpha}_{\delta} c_{\alpha} \partial^{\delta} \omega - b^{\alpha}_{\delta} c^{\delta} \partial_{\alpha} \omega\right)}_{=0 \text{ after index redefinition}},$$

so that $S'_{\text{GH}} = S_{\text{GH}}$.

Thanks to this property we can choose the fiducial metric to be $\hat{h}_{\alpha\beta} = e^{-\omega} \eta_{\alpha\beta}$ with an arbitrary $\omega$ and reduce the complete action to

$$S = -\frac{1}{4\pi\alpha'} \int d^2\sigma \, \partial_{\alpha} X^{\mu} \partial^{\alpha} X_{\mu} - \frac{i}{2\pi} \int d^2\sigma \, b_{\alpha\beta} \partial^{\alpha} c^{\beta}. \tag{3.239}$$

This action is already written as the reduced BRST action, associated to the reduced BRST transformation needed to evaluate the BRST charge. To find the action of $\delta_B$ over all the fields we must, as seen in section 2.2, evaluate the equations of motion of the metric to find the equations of motion of the Lagrange multiplier $B_{\alpha\beta}$ appearing in the anti-ghost BRST transformation $\delta_B b_{\alpha\beta} = iB_{\alpha\beta}$. To do so we must consider the extended BRST action

$$S_{\text{BRST}} = -\frac{1}{4\pi\alpha'} \int d^2\sigma \sqrt{-h} h^{\alpha\beta} \partial_{\alpha} X^{\mu} \partial_{\beta} X_{\mu} - \frac{i}{2\pi} \int d^2\sigma \sqrt{-h} b^{\alpha\beta} \nabla_{\alpha} c_{\beta}$$

$$- \frac{1}{4\pi} \int d^2\sigma \sqrt{-h} B_{\alpha\beta} (h^{\alpha\beta} - \hat{h}^{\alpha\beta}), \tag{3.240}$$

and evaluate

$$\frac{4\pi}{\sqrt{-h}} \frac{\delta S_{\text{BRST}}}{\delta h^{\alpha\beta}} = -B_{\alpha\beta} + T^M_{\alpha\beta} + T^{GH}_{\alpha\beta}, \tag{3.241}$$

where $T^M$ is the matter stress-energy tensor defined in (3.14), while $T^{GH}$ is the ghost stress energy tensor that we will now explicitly evaluate. To do so we consider action (3.237) and

we evaluate the variation:

$$\delta_h S_{GH} = -\frac{i}{2\pi} \int d^2\sigma \left[ -\frac{1}{2}\sqrt{-h}\, h_{\rho\sigma}\delta h^{\rho\sigma} h^{\mu\alpha} b_{\mu\beta}\nabla_\alpha c^\beta + \sqrt{-h}\delta h^{\mu\alpha} b_{\mu\beta}\nabla_\alpha c^\beta \right. \tag{3.242}$$
$$\left. + \sqrt{-h}\delta h^{\mu\alpha} b_{\mu\beta}\delta\Gamma^\beta_{\alpha\gamma}c^\gamma \right].$$

The variation of the Christoffel symbol as a function of the variation of the metric tensor can be written in the well known form

$$\delta\Gamma^\beta_{\alpha\gamma} = \frac{1}{2}h^{\beta\lambda}\left( \nabla_\gamma\delta h_{\lambda\alpha} + \nabla_\alpha\delta h_{\lambda\gamma} - \nabla_\lambda\delta h_{\alpha\gamma} \right), \tag{3.243}$$

which can be expressed with respect to the variation of the inverse metric tensor using the property

$$h_{\alpha\beta}h^{\beta\gamma} = \delta^\gamma_\alpha \rightarrow \delta h_{\alpha\beta}h^{\beta\gamma} + h_{\alpha\beta}\delta h^{\beta\gamma} = 0 \implies \delta h_{\alpha\beta} = -h_{\alpha\delta}h_{\beta\gamma}\delta h^{\delta\gamma}, \tag{3.244}$$

so that

$$\delta\Gamma^\beta_{\alpha\gamma} = -\frac{1}{2}h^{\beta\gamma}\left( h_{\lambda\mu}h_{\alpha\nu}\nabla_\gamma\delta h^{\mu\nu} + h_{\lambda\mu}h_{\gamma\nu}\nabla_\alpha\delta h^{\mu\nu} + h_{\alpha\mu}h_{\gamma\nu}\nabla_\lambda\delta h^{\mu\nu} \right). \tag{3.245}$$

Using this equation, the third term in (3.242) becomes

$$\delta h^{\mu\alpha} b_{\mu\beta}\delta\Gamma^\beta_{\alpha\gamma}c^\gamma = -\frac{1}{2}h^{\rho\alpha}h^{\beta\lambda}h_{\lambda\mu}h_{\alpha\nu}b_{\rho\beta}c^\gamma\nabla_\gamma\delta h^{\mu\nu} \tag{3.246}$$
$$-\frac{1}{2}h^{\rho\alpha}h^{\beta\lambda}h_{\lambda\mu}h_{\gamma\nu}b_{\rho\beta}c^\gamma\nabla_\alpha\delta h^{\mu\nu}$$
$$+\frac{1}{2}h^{\rho\alpha}h^{\beta\lambda}h_{\alpha\mu}h_{\gamma\nu}b_{\rho\beta}c^\gamma\nabla_\lambda\delta h^{\mu\nu},$$

that can be further simplified using the Leibniz rule and ignoring all the boundary terms to obtain

$$\delta h^{\mu\alpha} b_{\mu\beta}\delta\Gamma^\beta_{\alpha\gamma}c^\gamma = \frac{1}{2}\delta h^{\mu\nu}\left[ \nabla_\gamma(b_{\mu\nu}c^\gamma + \nabla_\alpha(b^\alpha_\mu c_\nu) - \nabla_\lambda(b^\lambda_\mu c_\nu) \right] + \text{boundary terms}$$
$$= \frac{1}{2}\delta h^{\mu\nu}\nabla_\gamma(b_{\mu\nu}c^\gamma) + \text{boundary terms}.$$

Using this result, we then rewrite the ghost action variation as

$$\delta_h S_{GH} = -\frac{i}{2\pi}\int d^2\sigma\sqrt{-h}\,\delta h^{\mu\nu}\left[ \frac{1}{2}\nabla_\gamma(b_{\mu\nu}c^\gamma) + b_{\mu\beta}\nabla_\nu c^\beta - \frac{1}{2}h_{\mu\nu}b^\alpha_\beta\nabla_\alpha c^\beta \right]$$
$$= -\frac{i}{4\pi}\int d^2\sigma\sqrt{-h}\,\delta h^{\mu\nu}\left[ \nabla_\gamma(b_{\mu\nu}c^\gamma) + b_{\mu\beta}\nabla_\nu + b_{\nu\beta}\nabla_\mu c^\beta - h_{\mu\nu}b^\alpha_\beta\nabla_\alpha c^\beta \right]$$
$$= -\frac{i}{4\pi}\int d^2\sigma\sqrt{-h}\left[ \delta h^{\mu\nu}b_{\mu\nu}\nabla_\gamma c^\gamma + \delta h^{\mu\nu}\nabla_\gamma b_{\mu\nu}c^\gamma \right]$$
$$- \frac{i}{4\pi}\int d^2\sigma\sqrt{-h}\,\delta h^{\mu\nu}\left[ b_{\mu\beta}\nabla_\nu c^\beta + b_{\nu\beta}\nabla_\mu c^\beta - h_{\mu\nu}b^\alpha_\beta\nabla_\alpha c^\beta \right]. \tag{3.247}$$

The term $\delta h^{\mu\nu}b_{\mu\nu}\nabla_\gamma c^\gamma$ is identically zero because $\delta h^{\mu\nu}b_{\mu\nu} = \delta(h^{\mu\nu}b_{\mu\nu}) = \delta b^\alpha_\alpha = 0$ and in the end we find

$$\delta_h S_{GH} = -\frac{i}{4\pi}\int d^2\sigma\sqrt{-h}\,\delta h^{\mu\nu}\left[ \nabla_\gamma b_{\mu\nu}c^\gamma + b_{\mu\beta}\nabla_\nu c^\beta + b_{\nu\beta}\nabla_\mu c^\beta - h_{\mu\nu}b^\alpha_\beta\nabla_\alpha c^\beta \right], \tag{3.248}$$

so that the symmetric ghost stress energy tensor becomes

$$T^{GH}_{\alpha\beta} = \frac{4\pi}{\sqrt{-h}} \frac{\delta S}{\delta h^{\alpha\beta}} = -i \left( \nabla_\gamma b_{\alpha\beta} c^\gamma + b_{\alpha\gamma} \nabla_\beta c^\gamma + b_{\beta\gamma} \nabla_\alpha c^\gamma - h_{\alpha\beta} b^\delta_\gamma \nabla_\delta c^\gamma \right). \tag{3.249}$$

If we impose the metric to be 2-dimensional flat Minkowski, we finally get the following value of the Lagrange multiplier:

$$B_{\alpha\beta} = \left[ T^M_{\alpha\beta} - i \left( \partial_\gamma b_{\alpha\beta} c^\gamma + b_{\alpha\gamma} \partial_\beta c^\gamma + b_{\beta\gamma} \partial_\alpha c^\gamma - \eta_{\alpha\beta} b^\gamma_\delta \partial_\gamma c^\delta \right) \right]. \tag{3.250}$$

To make contact with the old covariant quantization we choose the worldsheet light-cone coordinates. This means that the trace-less condition of the anti-ghost field becomes

$$b^\alpha_{\ \alpha} = \eta^{\alpha\beta} b_{\alpha\beta} = \eta^{+-} b_{+-} + \eta^{-+} b_{-+} = -4 b_{-+} = -4 b_{+-} = 0, \tag{3.251}$$

so that the only non zero anti-ghost terms are $b_{++}$ and $b_{--}$, associated with the ghost $c^+$ and $c^-$. In this set of coordinates the reduced BRST transformations can be written as

$$\delta_B X^\mu = c^+ \partial_+ X^\mu + c^- \partial_- X^\mu, \tag{3.252a}$$

$$\delta_B c^\pm = c^\pm \partial_\pm c^\pm, \tag{3.252b}$$

$$\delta_B b_{\pm\pm} = i \left( -\frac{1}{\alpha'} \partial_\pm X^\mu \partial_\pm X_\mu - i(2 b_{\pm\pm} \partial_\pm c^\pm + \partial_\pm b_{\pm\pm} c^\pm) \right). \tag{3.252c}$$

### 3.3.1 Canonical quantization

The action in the conformal gauge becomes

$$S = \frac{1}{2\pi\alpha'} \int d\sigma^+ d\sigma^- \partial_+ X^\mu \partial_- X_\mu - \frac{i}{2\pi} \int d\sigma^+ d\sigma^- (b_{++} \partial_- c^+ + b_{--} \partial_+ c^-), \tag{3.253}$$

with associated equations of motion:

$$\partial_+ \partial_- X^\mu = 0, \tag{3.254}$$

$$\partial_+ c^- = \partial_- c^+ = \partial_+ b_- = \partial_- b_+ = 0. \tag{3.255}$$

In the reduced BRST formalism the BRST transformation is a nilpotent symmetry of the action only when all the equations of motion are solved simultaneously. This is not a restriction because the canonical quantization is done on the space of solutions. Solving the equations of motion we find again the same matter fields of the old covariant quantization while for the ghost fields we find

$$c^+ = c(\sigma^+), \quad c^- = \tilde{c}(\sigma^-), \tag{3.256}$$

$$b^+ = b(\sigma^+), \quad b^- = \tilde{b}(\sigma^-). \tag{3.257}$$

Using the $2\pi$-periodicity of the string we can expand these fields using a Fourier series:

$$c(\sigma^+) = \sum_{n\in\mathbb{Z}} c_n e^{-in\sigma^+}, \qquad \tilde{c}(\sigma^-) = \sum_{n\in\mathbb{Z}} \tilde{c}_n e^{-in\sigma^-}, \tag{3.258}$$

$$b(\sigma^+) = \sum_{n\in\mathbb{Z}} b_n e^{-in\sigma^+}, \qquad \tilde{b}(\sigma^-) = \sum_{n\in\mathbb{Z}} \tilde{b}_n e^{-in\sigma^-}, \tag{3.259}$$

with the reality conditions $b_n^+ = b_{-n}$, $\tilde{b}_n^+ = \tilde{b}_{-n}$, $c_n^+ = c_{-n}$ and $\tilde{c}_n^+ = \tilde{c}_{-n}$. Because of the graded commutator relations (with $b$ being the canonically conjugated momentum of $c$)

$$[b(\tau,\sigma), c(\tau,\sigma')] = [\tilde{b}(\tau,\sigma), \tilde{c}(\tau,\sigma')] = 2\pi\delta(\sigma - \sigma'), \tag{3.260}$$

the expansion coefficients must also satisfy

$$[b_n, c_m] = \delta_{n+m}, \quad [\tilde{b}_n, \tilde{c}_m] = \delta_{n+m}, \tag{3.261}$$

$$[c_n, c_m] = 0, \quad [\tilde{b}_n, \tilde{b}_m] = 0. \tag{3.262}$$

It is easy to check that these relations actually reconstruct eq. (3.260):

$$[b(\tau, \sigma), c(\tau, \sigma')] = \sum_{n,m} [b_n, c_m] e^{-i(n\sigma + m\sigma')} = \sum_{n \in \mathbb{Z}} e^{-in(\sigma - \sigma')} = 2\pi\delta(\sigma - \sigma'). \tag{3.263}$$

We can now analyze the structure of the Hilbert space as already did for the free particle, namely we divide the total Hilbert space in matter and ghost sector:

$$\mathcal{H}_{\text{tot}} = \mathcal{H}_{\text{matter}} \otimes \mathcal{H}_{\text{ghost}} = \mathcal{H}_m \otimes \mathcal{H}_{gh}. \tag{3.264}$$

The natural way we can build the ghost sector is to consider ghost creation and annihilation operators on a ground state $|\downarrow\rangle$ annihilated by $b_n, c_n, n > 0$:

$$b_n|\downarrow\rangle = c_n|\downarrow\rangle = 0, \qquad \langle\downarrow|b_{-n} = \langle\downarrow|c_{-n} = 0, \quad \text{for } n > 0. \tag{3.265}$$

If $n = 0$ then $c_0|\downarrow\rangle = |\uparrow\rangle$, $b_0|\downarrow\rangle = 0$ with the following normalization:

$$\langle\downarrow|\uparrow\rangle = \langle\downarrow|c_0|\downarrow\rangle = 1, \tag{3.266a}$$

$$\langle\uparrow|\uparrow\rangle = \langle\downarrow|\downarrow\rangle = 0. \tag{3.266b}$$

The general ghost state will then be

$$b_{-n_1} \ldots b_{-n_N} c_{-m_1} \ldots c_{-m_M} |\downarrow\rangle, \quad n_i > 0, \ m_j \geq 0. \tag{3.267}$$

### 3.3.2 $SL(2, \mathbb{C})$ invariant vacuum and the ghost Virasoro algebra

The construction we have just carried out can be reformulated with a more suitable definition of the vacuum state. In particular we can then define a new vacuum state $|0\rangle$ so that

$$c_n|0\rangle = 0, \quad \forall n \geq 2, \tag{3.268a}$$

$$b_m|0\rangle = 0, \quad \forall m \geq -1. \tag{3.268b}$$

Why is this a better choice of vacuum state? We can understand this by evaluating the form of the stress-energy tensor of the ghost sector using the definitions (3.258), (3.259) of $c$ and $b$:

$$
\begin{aligned}
T_{++}^{(gh)} &= i(2\partial_+ c^+ b_{++} + c^+ \partial_+ b_{++}) \\
&= i \sum_{n,m \in \mathbb{Z}} 2(-in) c_n b_m e^{-i(n+m)\sigma^+} + i \sum_{n,m \in \mathbb{Z}} (-im) c_n b_m e^{-i(n+m)\sigma^+} \\
&= \sum_{n,m \in \mathbb{Z}} (2n + m) c_n b_m e^{-i(n+m)\sigma^+} = -\sum_{n,k \in \mathbb{Z}} (n+k) b_{k-n} c_n e^{-ik\sigma^+} \\
&= -\sum_{k \in \mathbb{Z}} \Big[ \sum_{n \in \mathbb{Z}} (k-n) b_{k+n} c_{-n} \Big] e^{-ik\sigma^+} = \sum_{k \in \mathbb{Z}} L_k^{(gh)} e^{-ik\sigma^+}. 
\end{aligned}
\tag{3.269}
$$

We have defined the *ghost sector Virasoro's* $L_k^{(gh)}$:

$$L_k^{(gh)} = \sum_{n \in \mathbb{Z}} (n-k) b_{k+n} c_{-n} \qquad \text{(classical)}. \tag{3.270}$$

In the quantum theory we normal order them on the new vacuum state $|0\rangle$:

$$L_k^{(gh)} = \sum_{n \in \mathbb{Z}} (n-k) : b_{k+n} c_{-n} : \qquad \text{(quantum)}. \qquad (3.271)$$

We will now prove that these operators satisfy the Virasoro algebra with central charge $c = -26$. At first we evaluate two fundamental commutation relations:

$$\begin{aligned}
[L_n^{(gh)}, b_m] &= \sum_{k \leq -2} (n-k)[b_{n+k} c_{-k}, b_m] - \sum_{k \geq -1} (n-k)[c_{-k} b_{n+k}, b_m] \\
&= \sum_{k \leq -2} (n-k) b_{n+k}[c_{-k}, b_m] + \sum_{k \geq -1} (n-k)[c_{-k}, b_m] b_{n+k} = (n-m) b_{n+m}, \quad (3.272a) \\
[L_n^{(gh)}, c_m] &= \sum_{k \leq -2} (n-k)[b_{n+k} c_{-k}, c_m] - \sum_{k \geq -1} (n-k)[c_{-k} b_{n+k}, c_m] \\
&= -\sum_{k \leq -2} (n-k) c_{-k} \delta_{n+m+k,0} = -(2n+m) c_{n+m}. \quad (3.272b)
\end{aligned}$$

Thanks to these we then obtain

$$\begin{aligned}
[L_n^{(gh)}, L_m^{(gh)}] &= \sum_{k \leq -2} (m-k)[L_n^{(gh)}, b_{m+k} c_{-k}] - \sum_{k \geq -1} (m-k)[L_n^{(gh)}, c_{-k} b_{m+k}] \\
&= \sum_{k \leq -2} (m-k)(n-m-k) b_{n+m+k} c_{-k} - \sum_{k \leq -2} (m-k)(2n-k) b_{m+k} c_{n-k} \\
&\quad + \sum_{k \geq -1} (m-k)(2n-k) c_{n-k} b_{m+k} - \sum_{k \geq -1} (m-k)(n-m-k) c_{-k} b_{n+m+k} \\
&= \sum_{k \leq -2} (m-k)(n-m-k) b_{n+m+k} c_{-k} - \sum_{q \leq -n-2} (m-n-q)(n-q) b_{m+n+q} c_{-q} \\
&\quad + \sum_{q \geq -n-1} (n-q)(m-n-q) c_{-q} b_{n+m+k} - \sum_{k \geq -1} (m-k)(n-m-k) c_{-k} b_{n+m+k} \\
&= \sum_{q \leq -n-2} \underbrace{[(m-q)(n-m-q) - (n-q)(m-n-q)]}_{(n-m)(n+m-q)} b_{q+m+n} c_{-q} \\
&\quad - \sum_{q \geq -1} [(m-q)(n-m-q) - (n-q)(m-n-q)] c_{-q} b_{q+m+n} \\
&\quad + \sum_{q=-n-1}^{-2} (m-q)(n-m-q) b_{q+m+n} c_{-q} + \sum_{q=-n-1}^{-2} \underbrace{(m-q)(n-m-q) c_{-q} b_{q+m+n}}_{\text{In this range this is not normal ordered}} \\
&= \sum_{-1 \leq q \leq -n-2} (n-m)(n+m-q) : b_{q+m+n} c_{-q} : + \sum_{q \leq -n-1}^{-2} (n-q)(m-n-q) \delta_{n+m} \\
&\quad + \sum_{q=-n-1}^{-2} [(m-q)(n-m-q) - (n-q)(m-n-q)] b_{q+m+n} c_{-q} \\
&= (n-m) \sum_{q \in \mathbb{Z}} (n+m-q) : b_{q+m+n} c_{-q} : -\delta_{m+n} \sum_{q=-n-1}^{-2} (n-q)(2n+q) \\
&= (n-m) L_{m+n}^{(gh)} - \frac{13}{6}(n^3-n) \delta_{n+m},
\end{aligned}$$

which gives rise to

$$[L_n^{(gh)}, L_m^{(gh)}] = (n-m) L_{m+n}^{(gh)} + \frac{-26}{12}(n^3-n) \delta_{n+m}. \qquad (3.273)$$

Here we find the same type of central extension $\frac{c}{12}(n^3 - n)\delta_{n+m}$ that we got in the matter sector, but this time the *central charge* has a specific universal value

$$c^{\text{ghost}} = -26\,. \tag{3.274}$$

It is easy now to verify that $L_{0,\pm 1}^{gh}|0\rangle = 0$, namely the vacuum state is annhiliated by the three Virasoros satisfying the $SL(2,\mathbb{C})$ algebra. This means that from the point of view of the ghost sector the new vacuum state is $SL(2,\mathbb{C})$ invariant. In reality because the same property is shared with the matter Virasoros this new vacuum state is globally $SL(2,\mathbb{C})$ invariant. For this reason $|0\rangle$ is known as the $SL(2,\mathbb{C})$ invariant vacuum state.

We can now try to understand the link between the new and old vacuum state. It is easy to understand that we can easily identify

$$|\downarrow\rangle = c_1 |0\rangle\,, \tag{3.275}$$

because $c_1|0\rangle$ shares all the requested properties with $|\downarrow\rangle$. Namely we can verify that

$$c_n|\downarrow\rangle = c_n c_1|0\rangle = 0\,, \quad \forall n > 0, \tag{3.276a}$$

$$c_0|\downarrow\rangle = c_0 c_1|0\rangle \neq 0 \implies |\uparrow\rangle = c_0 c_1|0\rangle\,, \tag{3.276b}$$

$$b_n|\downarrow\rangle = b_n c_1|0\rangle = 0\,, \quad \forall n \geq 0, \tag{3.276c}$$

$$\langle\uparrow|\uparrow\rangle = \langle\downarrow|c_0^2|\downarrow\rangle = 0, \tag{3.276d}$$

$$\langle\downarrow|\downarrow\rangle = \langle\downarrow|[b_0,c_0]|\downarrow\rangle = 0, \tag{3.276e}$$

$$\langle\downarrow|\uparrow\rangle = \langle 0|c_{-1}c_0 c_1|0\rangle = 1\,. \tag{3.276f}$$

In general we will find it convenient to use the following four vacuum states

$$|0\rangle\,, \qquad |\downarrow\rangle = c_1|0\rangle\,, \qquad |\uparrow\rangle = c_0 c_1|0\rangle\,, \qquad \big|\hat{0}\big\rangle = c_{-1}c_0 c_1|0\rangle\,. \tag{3.277}$$

All these states can be characterized using the *ghost number operator*, defined as:

$$J_{gh} = -\sum_{k\geq 2} b_{-k}c_k + \sum_{k\geq -1} c_{-k}b_k\,, \tag{3.278}$$

which counts a $-1$ for every anti-ghost operator $b_n$ and a $+1$ for every ghost operator $c_n$ in the state. With this definition we have that

$$J_{gh}|0\rangle = 0\,, \qquad J_{gh}|\downarrow\rangle = |\downarrow\rangle\,, \qquad J_{gh}|\uparrow\rangle = 2|\uparrow\rangle\,, \qquad J_{gh}\big|\hat{0}\big\rangle = 3\big|\hat{0}\big\rangle\,. \tag{3.279}$$

Any other states will be written as a linear combination of

$$\big|\{n_i\},\{m_j\}\big\rangle = c_{-n_1}\dots c_{-n_k} b_{-m_1}\dots b_{-m_q}|0\rangle\,, \tag{3.280}$$

$$J_{gh}\big|\{n_i\},\{m_j\}\big\rangle = (k-q)\big|\{n_i\},\{m_j\}\big\rangle\,, \quad \text{for } n_i \geq -1\,, \ m_j \geq 2\,. \tag{3.281}$$

### 3.3.3 BRST charge and critical dimension

We are now ready to construct the BRST charge of the string. Considering the general construction for a set of constraints, in this case we have

$$L_n \sim 0\,, \qquad \tilde{L}_m \sim 0\,. \tag{3.282}$$

The constraints satisfy, upon quantization, the (matter) Virasoro algebra:

$$[L_n, L_m] = (n-m)L_{n+m} + \frac{D}{12}n(n^2 - 1)\delta_{n+m}\,. \tag{3.283}$$

Following the standard construction in section 2.3.1 we identify $\mathcal{F}_i = L_n$ and the structure constants of the algebra with

$$f^k_{nm} = \delta_{k,n+m}(n-m). \tag{3.284}$$

Using (2.103) we obtain

$$\tilde{\mathcal{F}}_i = -f^k_{ij}c^j b_k = -\sum_{j,k}(i-j)\delta_{k,i+j}c^j b_k, \tag{3.285}$$

where $[b_k, c^j] = \delta^k_j$. To change these to our natural ghost operators we must redefine $c^i \to c_{-i}$ and $b_i \to b_i$ so that $[b_k, c_j] = \delta_{k+j}$, and the ghost generator becomes

$$\tilde{\mathcal{F}}_i = -\sum_{j,k}(i-j)\delta_{k,i+j}c_{-j}b_k = -\sum_{j,k}(i-j)c_{-j}b_{i+j} = \sum_j(i-j)b_{i+j}c_{-j} = L^{gh}_i. \tag{3.286}$$

Considering the normal ordering on the invariant vacuum, $\tilde{\mathcal{F}}_i = L^{gh}_i$. Using (2.109) the BRST charge then becomes

$$Q = c^i\mathcal{F}_i + \frac{1}{2}c^i\tilde{\mathcal{F}}_i = \sum_{n\in\mathbb{Z}}\left[c_{-n}L^{(m)}_n + \frac{1}{2}:c_{-n}L^{(gh)}_n:\right], \tag{3.287}$$

where we have already considered the normal ordering in the quantum case.

Alternatively, this BRST charge can be constructed as the Noether charge associated to the BRST transformations (3.252a), such that $\delta_B = i[\cdot, Q_B]$.

Defining now the total Virasoro operators as $L^{(tot)}_n = L^{(m)}_n + L^{(gh)}_n$ obeying the total Virasoro algebra as

$$[L^{(tot)}_n, L^{(tot)}_m] = (n-m)L^{(tot)}_{n+m} + \frac{D-26}{12}n(n^2-1)\delta_{n+m}, \tag{3.288}$$

it is possible to prove that this BRST charge satisfies

$$Q^2 = \frac{1}{2}[Q,Q] = \frac{1}{2}\sum_{n,m}\left[[L^{(tot)}_n, L^{(tot)}_m] - (n-m)L^{(tot)}_{n+m}\right]c_{-n}c_{-m} = 0, \tag{3.289}$$

which means that the BRST charge is nilpotent if and only if $D = 26$. This calculation is rather tedious but in principle straightforward once normal ordering subtleties are correctly taken into account. It is also possible to simply and elegantly prove that $Q^2 = 0 \Rightarrow D = 26$.

*Proof*
Using the property

$$[Q, b_n] = \mathcal{F}^{tot}_n = L^{(tot)}_n, \tag{3.290}$$

we can prove that $[Q, L^{(tot)}_n] = 0$:

$$[Q, L^{(tot)}_n] = [Q, [Q, b_n]] = [[Q,Q], b_n] - [Q, [Q, b_n]] \to 2[Q, L^{(tot)}_n] = [Q^2, b_n] = 0. \tag{3.291}$$

If we consider this identity in the commutation relation of two total Virasoros we get

$$[L^{(tot)}_n, L^{(tot)}_m] = [L^{(tot)}_n, [Q, b_m]] = [[L^{(tot)}_n, Q], b_m] + [Q, [L^{(tot)}_n, b_m]] \tag{3.292}$$

$$= [Q, [L^{(gh)}_n, b_m]] = (n-m)[Q, b_{n+m}] = (n-m)L^{(tot)}_{n+m}, \tag{3.293}$$

which means that $c_{tot} = 0$ that is $D = 26$.

### 3.3.4 String spectrum from the BRST charge

In this section we will describe the physical cohomology in (say) the left moving sector. An analogous mirror story can be repeated for the right-moving sector. The left-moving results will directly apply to the open string spectrum, modulo possible boundary details which will not be important in this section. As we saw the total BRST Hilbert space can be decomposed into the direct product of matter and ghost sector $\mathcal{H} = \mathcal{H}^m \otimes \mathcal{H}^{gh}$:

- $\mathcal{H}^m \rightarrow [\alpha_n^\mu, \alpha_m^\nu] = \eta^{\mu\nu} n \delta_{n+m}, \qquad [\hat{X}^\mu, \hat{P}^\nu] = i\eta^{\mu\nu},$
- $\mathcal{H}^{gh} \rightarrow [b_n, c_m] = \delta_{n+m}.$

Inside this total space we can define four 'vacua' with definite momentum:

$$|0, P\rangle = e^{i\hat{X}\cdot P} |0\rangle, \tag{3.294a}$$

$$|\downarrow, P\rangle = e^{i\hat{X}\cdot P} |\downarrow\rangle = e^{i\hat{X}\cdot P} c_1 |0\rangle, \tag{3.294b}$$

$$|\uparrow, P\rangle = e^{i\hat{X}\cdot P} |\uparrow\rangle = e^{i\hat{X}\cdot P} c_0 c_1 |0\rangle, \tag{3.294c}$$

$$|\hat{0}, P\rangle = e^{i\hat{X}\cdot P} |\hat{0}\rangle = e^{i\hat{X}\cdot P} c_{-1} c_0 c_1 |0\rangle. \tag{3.294d}$$

The normalization is chosen so that

$$\langle 0, P' | c_{-1} c_0 c_1 | 0, P\rangle = \delta(P - P'). \tag{3.295}$$

This setting is similar to the configuration we have found in the case of the free particle where the global space was spanned by $|\downarrow, P\rangle \oplus |\uparrow, P\rangle$. However here in the string case we have also all possible positive and negative ghost numbers and in particular two additional ghost vacua ($|0\rangle$ and $|\hat{0}\rangle$) that are useful. All the other states in the string Hilbert space $\mathcal{H}$ are then obtained applying different oscillators to these vacuums. Of course it is up to us to choose which vacuum to use, because all vacua are obtainable from one another by using the ghost zero modes $(c_{-1}, c_0, c_1)$ and $(b_{-1}, b_0, b_1)$.

On $\mathcal{H}$ there is a nilpotent BRST charge which acts as an operator:[16]

$$Q = \sum_{n\in\mathbb{Z}} \left( c_n L_{-n} + \frac{1}{2} : c_n L_{-n}^{(gh)} : \right), \tag{3.296}$$

with $: \cdot :$ the normal ordering with respect to $|0\rangle$ and $L_n$ and $L_n^{(gh)}$ the matter and ghost Virasoros:

$$L_n = \frac{1}{2} \sum_{k\in\mathbb{Z}} : \alpha_{n-k} \alpha_k :, \qquad L_n^{(gh)} = \sum_{k\in\mathbb{Z}} (n-k) : b_{n+k} c_{-k} :, \tag{3.297}$$

$$[L_n, L_m] = (n-m) L_{n+m} + \frac{D}{12} (n^3 - n) \delta_{n+m}, \tag{3.298}$$

$$[L_n^{(gh)}, L_m^{(gh)}] = (n-m) L_{n+m}^{(gh)} + \frac{-26}{12} (n^3 - n) \delta_{n+m}. \tag{3.299}$$

We are now ready to obtain the physical states $|phys\rangle \in \mathcal{H}$ analyzing the cohomology of $Q$. We start with $\#_{gh} < 0$ and we only state the following result.

*Theorem*
Every closed state at negative ghost number is also exact, namely there is no cohomology for $\#_{gh} < 0$. The same is true for $\#_{gh} > 3$.

Therefore the cohomology is concentrated at ghost number 0,1,2,3.

---

[16]We omit the matter symbol on matter Virasoro operators from now on.

■ $\#_{gh} = 0$

The fundamental state at this ghost number is the zero-momentum vacuum state. It is easy to prove that $Q|0\rangle = 0$ given the normal ordering of $Q$, but we now must see if $|0\rangle$ is exact, namely if $\exists |\Lambda\rangle \mid Q|\Lambda\rangle = |0\rangle$. If we define the total level $N^{tot} = L_0^{(tot)} - \frac{1}{2}\alpha_0^2$ we have

$$[Q, N^{(tot)}] = [Q, L_0^{(tot)}] - \frac{1}{2}[Q, \alpha_0^2] = 0, \tag{3.300}$$

which means that the BRST charge does not modify the level of a state. Using this result we can conclude that $|\Lambda\rangle$ must be at zero level and with ghost number $-1$. The only state with this property would be $|\Lambda\rangle = b_0|0\rangle$ which is, however, identically zero from the definition of the vacuum state. This means that $\nexists |\Lambda\rangle$ and $|0\rangle$ is in the cohomology.

It can be shown that $|0\rangle$ is the only element of the cohomology at ghost number zero.

■ $\#_{gh} = 1$

At the ghost number 1 we will search the cohomology in the subclass of states $|\psi\rangle = c_1|\phi_m\rangle = |\phi_m\rangle \otimes |\downarrow\rangle$ where $|\phi_m\rangle$ represents a generic matter state. These states obey the so-called Siegel gauge condition $b_0|\psi\rangle = 0$.[17] If we apply $Q$ to this state we obtain

$$Q|\psi\rangle = Qc_1|\phi_m\rangle = [Q, c_1]|\phi_m\rangle - c_1 Q|\phi_m\rangle. \tag{3.301}$$

This can be further manipulated using the property

$$[Q, c_n] = -\sum_{m\in\mathbb{Z}}(m+2n)c_{-m}c_{n+m}, \tag{3.302}$$

so that

$$\begin{aligned}
[Q, c_1]|\phi_m\rangle &= -|\phi_m\rangle \otimes \sum_{m\in\mathbb{Z}}(m+2)c_{-m}c_{1+m}|0\rangle_{gh} \\
&= -|\phi_m\rangle \otimes (\dots - c_3 c_{-2} + c_1 c_0 + 2c_0 c_1 + 3c_{-1}c_2 + \dots)|0\rangle_{gh} \\
&= -|\phi_m\rangle \otimes (-c_1 c_0)|0\rangle_{gh} = c_1 c_0|\phi_m\rangle.
\end{aligned} \tag{3.303}$$

Thanks to this result we obtain

$$\begin{aligned}
Q|\psi\rangle &= c_1 c_0|\phi_m\rangle - c_1 \sum_{n\in\mathbb{Z}} c_n L_{-n}|\phi_m\rangle \\
&= c_1 \left( c_0 + \underbrace{\dots - c_2 L_{-2}}_{\text{All zero over } |\phi_m\rangle} - c_1 L_{-1} - c_0 L_0 - \sum_{n\geq 1} c_{-n} L_n \right)|\phi_m\rangle \\
&= (L_0 - 1)c_0 c_1|\phi_m\rangle + \sum_{n\geq 1} c_{-n} L_n c_1|\phi_m\rangle = 0,
\end{aligned} \tag{3.304}$$

which is zero if and only if

$$(L_0 - 1)|\phi_m\rangle = 0, \tag{3.305}$$

and

$$L_n|\phi_m\rangle = 0, \quad \forall n \geq 1, \tag{3.306}$$

which are the same physicality conditions we have obtained in the old covariant quantization. Therefore the BRST quantization implies the OCQ constraints with $a = 1$. The $(L_0^{(m)} - 1)$ should

---

[17]A consequence of this gauge condition is that the cohomology can be systematically searched for in the kernel of $L_0^{tot}$ since $b_0|\psi\rangle = Q|\psi\rangle = 0$ implies $[Q, b_0]|\psi\rangle = L_0^{tot}|\psi\rangle = 0$.

be thought of $L_0^{(\text{tot})} = L_0^{(\text{m})} + L_0^{(\text{gh})}$ where $L_0^{(\text{gh})} = -1$ when it acts on $c_1 |0\rangle = |\downarrow\rangle$. Therefore *the cohomology of $Q_B$ is properly contained in the kernel of $L_0^{(\text{tot})}$.*

In the old covariant quantization we have also defined the *null* states as those vectors satisfying

$$(L_0 - 1)|\text{null}\rangle = 0, \qquad L_{n \geq 1}|\text{null}\rangle = 0, \qquad |\text{null}\rangle = \sum_{n \geq 1} L_{-n}|\chi_n\rangle. \qquad (3.307)$$

OCQ null states correspond to BRST exact states. To see this consider the example of the open string gauge field in the BRST approach:

$$|\psi\rangle = A_\mu(P)\alpha_{-1}^\mu c_1 |0, P\rangle. \qquad (3.308)$$

We can add to it an exact state of the form

$$Q\lambda(P)|0, P\rangle = \sum_{n \in \mathbb{Z}} c_{-n} L_n |0, P\rangle = \lambda(P)(\underbrace{\ldots + c_2 L_{-2}}_{\text{Identically zero}} + c_1 L_{-1} + \underbrace{c_0 L_0 + \ldots}_{\text{zero when } P^2 = 0})|0, P\rangle$$

$$= \lambda(P)c_1 L_{-1}|0, P\rangle \sim P_\mu \lambda(P)\alpha_{-1}^\mu c_1 |0, P\rangle. \qquad (3.309)$$

We have already observed in (3.203) that when $P^2 = 0$ then $L_{-1}|0, P\rangle$ is a null state. Notice that this exact state still obeys the gauge condition $b_0 = 0$ which therefore does not fix the gauge completely, inside the cohomology.[18]

In general we have the property

$$c_1 |\text{null}\rangle = Q |\Lambda\rangle. \qquad (3.310)$$

In the end we can write the identifications

$$|\phi_m\rangle \longrightarrow c_1 |\phi_m\rangle = |\psi\rangle, \qquad (3.311a)$$

$$|\text{phys}\rangle \longrightarrow Q |\psi\rangle = 0, \qquad (3.311b)$$

$$|\text{null}\rangle \longrightarrow c_1 |\text{null}\rangle = Q |\Lambda\rangle, \qquad (3.311c)$$

and map the entire old covariant quantization in the ghost number one sector obeying the Siegel gauge condition $b_0 |\psi\rangle = 0$. It is also true, although we don't give a proof, that there is nothing else in the cohomology at ghost number one, only the (non null) OCQ physical states.

As an additional remark we can also notice that the parameter $a$ we have previously fixed to 1 is nothing but (minus) the eigenvalue of the state $c_1 |0\rangle$ under the action of $L_0^{(\text{gh})}$.

### ■ $\#_{gh} = 2$

Since at ghost number one we have restricted to states whose ghost content was entirely given by $|\downarrow\rangle = c_1 |0\rangle$, we can consider states at ghost number two of the form

$$|\hat{\psi}\rangle = c_0 c_1 |\phi_m\rangle = |\phi_m\rangle \otimes |\uparrow\rangle,$$

and apply the BRST charge to obtain

$$Q|\hat{\psi}\rangle = \underbrace{[Q, c_0 c_1]}_{=0}|\phi_m\rangle + c_0 c_1 Q |\phi_m\rangle$$

$$= c_0 c_1 (\ldots + c_1 L_{-1} + c_0 L_0 + \sum_{n \geq 1} c_{-n} L_n)|\phi_m\rangle = 0$$

$$\implies L_n |\phi_m\rangle = 0, \quad \forall n \geq 1. \qquad (3.312)$$

---

[18]This is analogous to Lorentz gauge $\partial_\mu A^\mu = 0$ which is preserved by $\delta A_\mu = \partial_\mu \lambda$ with $\Box \lambda = 0$.

Notice that this time we do not find the mass-shell condition $(L_0 - 1)|\phi_m\rangle = 0$. This situation is similar to what we have already encountered in the free particle model, where all the states $|P, \uparrow\rangle$ where in the kernel of $Q$, without any restriction on $P$. In that case the mass-shell condition was the only constraint on the particle state and it was recovered for $|P, \uparrow\rangle$ by studying the non-exactness condition. We must do the same for the string states in

$$\ker(Q)\Big|_{\text{gh}=2} = \left\{ |\hat{\psi}^*\rangle = c_0 c_1 |\phi_m^*\rangle \right\}_{L_{n\geq 1}|\phi_m^*\rangle = 0} . \tag{3.313}$$

We proceed by considering the states at ghost number one $|\psi^*\rangle = c_1 |\phi_m^*\rangle$ with $L_n |\phi_m^*\rangle = 0$, $\forall n \geq 1$ but not on-shell.

By definition we have the following identity:

$$b_0 |\hat{\psi}^*\rangle = c_1 |\phi_m^*\rangle = |\psi^*\rangle . \tag{3.314}$$

We can use it to obtain

$$Q|\psi^*\rangle = Q b_0 |\hat{\psi}^*\rangle = [Q, b_0]|\hat{\psi}^*\rangle - \underbrace{b_0 Q|\hat{\psi}^*\rangle}_{=0} = L_0^{(\text{tot})}|\hat{\psi}^*\rangle . \tag{3.315}$$

We now have two different possibilities.

1. If $L_0^{(\text{tot})}|\hat{\psi}^*\rangle \neq 0$ then, if we suppose to work on eigenstates of $L_0^{(\text{tot})}$ we can write

$$|\hat{\psi}^*\rangle = \frac{1}{L_0^{(\text{tot})}} Q|\psi^*\rangle = Q\left[ \frac{1}{L_0^{(\text{tot})}} |\psi^*\rangle \right] = Q\left[ \frac{b_0}{L_0^{(\text{tot})}} |\hat{\psi}^*\rangle \right], \tag{3.316}$$

and $|\hat{\psi}^*\rangle$ is exact, with $\frac{b_0}{L_0^{(\text{tot})}}|\hat{\psi}^*\rangle$ as trivializing state.

2. If $L_0^{(\text{tot})}|\hat{\psi}^*\rangle = 0$ then we have that the states are closed and not exact. Explicitly this means that

$$0 = L_0^{(\text{tot})}|\hat{\psi}^*\rangle = L_0^{(\text{tot})} c_0 c_1 |\phi_m^*\rangle = (L_0 + L_0^{(\text{gh})}) c_0 c_1 |\phi_m^*\rangle$$
$$= c_0 c_1 (L_0 - 1)|\phi_m^*\rangle , \tag{3.317}$$

and we again obtain that for the states to be physical they must also satisfy the mass-shell condition.

■ $\#_{gh} = 3$

In the ghost number 3 sector we focus on the state $|\hat{0}\rangle$ conjugate to the vacuum state $|0\rangle$ to prove that it is actually in the cohomology. At first we must prove $Q|\hat{0}\rangle = 0$:

$$Q|\hat{0}\rangle = \underbrace{\sum_{k \in \mathbb{Z}} (k-2) c_{-k} c_{k-1} c_0 c_1 |0\rangle}_{=0 \text{ for the properties of } c_l|0\rangle} - c_{-1} Q c_0 c_1 |0\rangle = c_{-1} Q^2 c_1 |0\rangle = 0 . \tag{3.318}$$

At the same time it must also not be exact but this is immediate since we known that

$$\langle 0|\hat{0}\rangle = \langle 0| c_{-1} c_0 c_1 |0\rangle = 1 , \tag{3.319}$$

which would not hold if $|\hat{0}\rangle = Q|\Lambda\rangle$ because $\langle 0|\hat{0}\rangle = \langle 0|Q|\Lambda\rangle = 0$.

All these states we have encountered in the cohomology have the property of having a positive definite scalar product as enunciated by the *no ghost theorem*.

SciPost Phys. Lect. Notes 90 (2025)

*No ghost theorem*

If we define

$$\left|\phi_i^{OCQ}\right\rangle \otimes \left|\downarrow\right\rangle = \left|\psi_i\right\rangle \ \text{at} \ \#_{gh} = 1, \tag{3.320}$$

$$\left|\phi_i^{OCQ}\right\rangle \otimes \left|\uparrow\right\rangle = \left|\hat{\psi}_i\right\rangle \ \text{at} \ \#_{gh} = 2, \tag{3.321}$$

satisfying $(L_n - \delta_{n,0})\left|\phi_i^{OCQ}\right\rangle = 0$ for $n \geq 0$, then

$$\langle\psi_i|\hat{\psi}_j\rangle = G_{ij}, \tag{3.322}$$

is positive semi-definite and become positive definite if we quotient-out the null states, or equivalently if we stay in the cohomology.

Therefore, in the cohomology, we only find positive norm states, which are consistent with Quantum Mechanics.

### 3.3.5 The (open) string field theory action

In the case of the free particle (2.98), once we had a well defined BRST charge and a non-degenerate inner product, it was possible to define a space-time action for the dynamical fields of the theory. Let us now focus on the open string on a D25 brane for concreteness. As we have seen, the physical states can be searched in the ghost number one sector of the Hilbert space. We thus select the dynamical string field to be $|\Psi\rangle$ with $\#_{gh}|\Psi\rangle = 1$, with a general expression

$$|\Psi\rangle = \int d^{26}P \left( t(P)c_1 e^{i\hat{X}\cdot P}|0\rangle + A_\mu(P)c_1\alpha_{-1}^\mu e^{i\hat{X}\cdot P}|0\rangle + iB(P)c_0 e^{i\hat{X}\cdot P}|0\rangle + \dots \right). \tag{3.323}$$

This state is physical if it satisfies

$$Q|\Psi\rangle = 0. \tag{3.324}$$

This is the equation of motion that extremizes the *free Open String Field Theory action*:

$$S[\Psi] = \frac{1}{2}\langle\Psi|Q|\Psi\rangle. \tag{3.325}$$

This action has also a local (off-shell) gauge symmetry

$$|\Psi\rangle \sim |\Psi\rangle + Q|\Lambda\rangle, \tag{3.326}$$

which generalizes the cohomological structure of the physical open string spectrum.

**Exercise 3.3.1**

Verify that given

$$|\Psi\rangle = \int d^{26}P \, t(P)c_1|0,P\rangle, \quad t \in \mathbb{R}, \tag{3.327}$$

the free string field action is equal to a scalar field action:

$$S[t] = \frac{\alpha'}{2} \int d^{26}P \, t(P^2 + m^2)t, \quad \text{with} \quad m^2 = -\frac{1}{\alpha'}. \tag{3.328}$$

Also verify that if we consider:

$$|\Psi\rangle = \int d^{26}P \left( A_\mu(P)c_1\alpha_{-1}^\mu + iB(P)c_0 \right)|0,P\rangle, \quad A_\mu, B \in \mathbb{R}, \tag{3.329}$$

we get the action of a $U(1)$ vector field coupled to an auxiliary field $B$. Integrate out $B$ and recover the covariant action of electromagnetism $\sim \int F_{\mu\nu}F^{\mu\nu}$, where $F_{\mu\nu} = \partial_\mu A_\nu - \partial_\nu A_\mu$.

The free action can be modified to include interactions, thanks to the addition of a 'simple' cubic vertex (the so-called Witten vertex)

$$S[\Psi] = \frac{1}{2}\langle \Psi, Q\Psi \rangle + \frac{1}{3}\langle \Psi, \Psi, \Psi \rangle. \tag{3.330}$$

With interactions turned one, the possibility of considering off-shell states allows us to move away from the perturbative vacuum of the theory, which is unstable due to the tachyon, to search for a stable vacuum if it exists. The presence of this true vacuum state has been conjectured by Ashoke Sen in 1999 and later analytically discovered by Martin Schnabl in 2005. It describes a configuration in which an initial unstable $D$-brane has decayed to nothing, leaving behind only pure spacetime and closed string radiation.

What about closed strings? One can try to proceed analogously to the open string case, taking the closed string field $|\Psi_c\rangle$ to be at total ghost number 2. However the natural expression $\langle \Psi_c|(Q_B + \bar{Q}_B)|\Psi_c\rangle$ is identically vanishing because it does not soak up the total ghost number six of the bracket. The way to resolve this problem is to *constrain* the off-shell space of closed string fields by asking for the (generalized) level matching conditions

$$(b_0 - \bar{b}_0)|\Psi_c\rangle = 0, \tag{3.331a}$$
$$(L_0 - \bar{L}_0)|\Psi_c\rangle = 0. \tag{3.331b}$$

In this restricted space there is a non-degenerate inner product given by

$$\langle \Psi_1, \Psi_2 \rangle \equiv \frac{1}{2}\langle \Psi_1|(c_0 - \bar{c}_0)|\Psi_2 \rangle. \tag{3.332}$$

With this inner product, calling $Q_{\text{tot}} = (Q_B + \bar{Q}_B)$, one can write down a consistent free action for closed string fields as

$$S[\Psi_c] = \frac{1}{2}\langle \Psi_c, Q_{\text{tot}}\Psi_c \rangle. \tag{3.333}$$

As for the open string field theory this free theory too can be deformed by adding interactions. However this time a cubic vertex is not sufficient to reproduce closed string amplitudes. It turns out that infinite interaction vertices are needed and these vertices are organized in a structure which is called $L_\infty$-algebra. As of today, this non-polynomial action is so complicated that it is not possible to ascertain whether there is a stable closed string tachyon vacuum, similar to the open string one.[19]

## 3.4 Lightcone quantization

At the classical level, choosing the conformal gauge and using worldsheet lightcone coordinates (3.43), we wrote (see eq. (3.48))

$$S[X] = \frac{1}{2\pi\alpha'}\int_{\text{WS}} d^2\sigma\, \partial_+ X^\mu \partial_- X_\mu. \tag{3.334}$$

---

[19]The interested student can find a detailed pedagogical introduction to String Field Theory in the recent book by H. Erbin, "String Field Theory: A Modern Introduction," Lect. Notes Phys. **980** (2021), 1-421 2021, [arXiv:2301.01686 [hep-th]].

As already noticed, this action is still invariant under the conformal diffeomorphisms (see eq. (3.49))

$$\sigma^+ \longrightarrow f_+(\sigma^+), \tag{3.335a}$$

$$\sigma^- \longrightarrow f_-(\sigma^-), \tag{3.335b}$$

which reflect on

$$X_L^\mu(\sigma^+) \longrightarrow X_L^\mu(f_+(\sigma^+)), \tag{3.336a}$$

$$X_R^\mu(\sigma^-) \longrightarrow X_R^\mu(f_-(\sigma^-)). \tag{3.336b}$$

This is a residual gauge redundancy and the Virasoro operators $L_n$ are precisely the generators of this residual gauge symmetry, as we will see in detail in the next chapter.

### 3.4.1  Quantization

In the previous chapter we have imposed the Virasoro constraints by using the BRST charge. Now we would like to do something easier: impose the constraints directly on the space of solutions for $X$ and then quantize the constrained set of solutions. In order to do so we go to lightcone coordinates *in space-time*

$$X^\pm = \frac{1}{\sqrt{2}}\left(X^0 \pm X^{D-1}\right). \tag{3.337}$$

Correspondingly we divide the spacetime directions into

$$\mu = (\underbrace{+,-}_{\substack{0,D-1\\ \text{(lightcone)}}}, \underbrace{i}_{1,\dots,D-2}), \tag{3.338}$$

where $i = 1,\dots,D-2$ are the transverse euclidean directions, while $+,-$ are the lightcone directions.

The idea of the lightcone quantization is then to *fix* the functional dependence of $X^+$ on $\sigma^\pm$ to be of the form

$$X_L^+(\sigma^+) = \frac{x_0^+ + c_0^+}{2} + \frac{\alpha'}{2}P^+\sigma^+, \qquad X_R^+(\sigma^-) = \frac{x_0^+ - c_0^+}{2} + \frac{\alpha'}{2}P^+\sigma^-, \tag{3.339a}$$

so that

$$X^+ = x_0^+ + \alpha' P^+ \tau. \tag{3.340}$$

This fixing is equivalent to stating that all the non-zero vibrational modes of the string along the $+$ direction are set to zero:

$$\alpha_{n\neq 0}^+ = 0. \tag{3.341}$$

After this choice for $X^+$ we are no more free to perform the conformal transformations (3.336a) because $X^+$ would not remain in the form (3.340).[20] If we perform a conformal transformation, we would reintroduce an oscillator dependence, by simply Fourier expanding $f_\pm(\sigma^\pm)$. Therefore we learn that $\alpha^+$ oscillators are not physical but are just a redundancy.

---

[20]In fact we are still free to perform constant translations in $(\sigma^+, \sigma^-)$ as we will discuss shortly.

The classical (i.e. before canonical quantization) Virasoro constraints are now given by (recalling eq. (3.341))

$$L_n = \frac{1}{2}\sum_{k\in\mathbb{Z}}\left(-2\alpha^-_{n-k}\alpha^+_k + \alpha^i_{n-k}\alpha^i_k\right) = -\alpha^-_n\alpha^+_0 + \frac{1}{2}\sum_{k\in\mathbb{Z}}\alpha^i_{n-k}\alpha^i_k. \qquad (3.342)$$

The idea now is to impose the $L_{n\neq 0} = 0$ constraints already at the classical level. This allows then to express the $\alpha^-_n$ oscillators in terms of the transverse oscillators

$$\alpha^-_n = \frac{1}{2\alpha^+_0}\sum_{k\in\mathbb{Z}}\alpha^i_{n-k}\alpha^i_k. \qquad (3.343)$$

In this game $L_0$ is treated differently: $L_0$ (and $\tilde{L}_0$) generate constant translations in the $\sigma^+, \sigma^-$ surface and these transformations don't change the generic form of $X^+$ because they don't introduce $\alpha^+_n$ oscillators and they don't change the momentum $P^+$. Therefore this will remain as a gauge redundancy which should be imposed on states after canonical quantization. This is in fact completely analogous to the particle and the associated (gauge) invariance of the gauge fixed world-line under constant translations in $\tau$.

There are only $D-2$ oscillators left, since $\alpha^+_{n\neq 0} = 0$ and $\alpha^-_n$ has been written in terms of $\alpha^i_n$. Therefore when we quantize we write

$$\left[\alpha^i_n, \alpha^j_m\right] = n\delta^{ij}\delta_{n+m,0}, \quad i,j = 1,\ldots,D-2. \qquad (3.344)$$

Inside this transverse Hilbert space we find states of the type

$$\alpha^{i_1}_{-n_1}\ldots\alpha^{i_k}_{-n_k}|0,P\rangle. \qquad (3.345)$$

On these states the only constraint to be imposed is $L_0$ (and $\tilde{L}_0$), whose classical expression is given by

$$L_0 = \frac{1}{2}(\alpha_0)^2 + \frac{1}{2}\sum_{k\neq 0}\alpha^i_{-k}\alpha^i_k. \qquad (3.346)$$

As usual we define the quantum mechanical operator to be normal ordered

$$L_0 = \frac{1}{2}(\alpha_0)^2 + \sum_{k\geq 1}\alpha^i_{-k}\alpha^i_k \qquad \text{(quantum)}, \qquad (3.347)$$

and then define the lightcone (lc) level operator as

$$N^{(\text{lc})} = \sum_{k\geq 1}\alpha^i_{-k}\alpha^i_k. \qquad (3.348)$$

The quantum mass-shell constraint (to be imposed on the states) will thus be

$$L_0 - a = \frac{1}{2}(\alpha_0)^2 + N^{(\text{lc})} - a = 0, \qquad (3.349)$$

where, following our usual arguments, $a$ is a to-be-determined normal ordering constant.

### 3.4.2 Spectrum

We will now analyze the spectrum of open strings on a D25-brane. The story for closed strings is analogous after the usual doubling of degrees of freedom. Going level by level we obtain what follows.

■ **Level $N^{(\text{lc})} = 0$**

At the level $N^{(\text{lc})} = 0$ we have the tachyon

$$t(P)|0, P\rangle \,. \tag{3.350}$$

Since we are considering an open string on a $D(25)$-brane we have $\alpha_0 = \sqrt{2\alpha'}P$ and therefore the constraint $L_0 - a = 0$ turns into

$$\alpha' m^2 = -a \,. \tag{3.351}$$

■ **Level $N^{(\text{lc})} = 1$**

At the level $N^{(\text{lc})} = 1$ the generic state is

$$A_i(P)\alpha_{-1}^i |0, P\rangle \,, \tag{3.352}$$

where $A_i(P)$ is a vector field in the vector irrep of $SO(D-2)$, (since $i = 1, \ldots, D-2$ are the only directions where we have oscillators). The constraint $L_0 - a = 0$ turns into

$$\alpha' m^2 = 1 - a \,. \tag{3.353}$$

This state is evidently a space-time vector which however has only $D - 2$ degrees of freedom. It is well known from quantum field theory that if a spacetime vector has only $D - 2$ physical degrees of freedom (per space-time point) then it should be massless. In this case there are indeed $D - 2$ physical polarizations which rotate according to $SO(D-2)$, the little Lorentz group of a light-like (i.e. massless) momentum $P$. Thus this state at $N^{(\text{lc})} = 1$ should be massless

$$\alpha' m^2 = 0 \,, \tag{3.354}$$

and this fixes

$$a = 1 \,, \tag{3.355}$$

exactly as in BRST (although for quite different conceptual reasons!).

■ **Level $N^{(\text{lc})} = 2$**

At the level $N^{(\text{lc})} = 2$ the generic state is

$$\left(\xi_i \alpha_{-2}^i + \xi_{ij} \alpha_{-1}^i \alpha_{-1}^j\right)|0, P\rangle \,, \tag{3.356}$$

with (setting $a = 1$ as we have just learned)

$$\alpha' m^2 = 1 \,. \tag{3.357}$$

We have that

- $\xi_i$ is in the vector representation of $SO(D-2)$, having $D - 2$ independent components. Using the Young tableaux notation, it can be written as $(\square)_{SO(D-2)}$.

- $\xi_{ij}$ is in the symmetric representation of $SO(D-2)$, having $\frac{1}{2}(D-1)(D-2)$ independent components. Using the Young tableaux notation, it can be written as $(\square\square)_{SO(D-2)}$ (this is still not an irrep of $SO(D-2)$ because the trace is not vanishing).

In total these two representations add up to provide the symmetric traceless representation of $SO(D-1)$ (the little group of a time-like (i.e. massive) momentum $P$) that we have found in OCQ:

$$(\square + \square\square^{\text{traceful}})_{SO(D-2)} = (\square\square^{\text{traceless}})_{SO(D-1)} \,, \tag{3.358}$$

as we can easily check by matching the degrees of freedom

$$(D-2) + \frac{1}{2}(D-2)(D-1) = \frac{1}{2}D(D-1) - 1 \,. \tag{3.359}$$

Therefore massive particles (irrepses of $SO(D-1)$) are reconstructed by irrepses of $SO(D-2)$, which is the only space-time symmetry which is manifest in the lightcone.

### 3.4.3 Zero point energy and renormalization

When you studied the harmonic oscillator in quantum mechanics you started with a Hamiltonian

$$\hat{H} = \frac{1}{2}\left(\hat{P}^2 + \omega^2 \hat{x}^2\right)$$
$$= \left(a^\dagger a + a a^\dagger\right) = a^\dagger a + \frac{1}{2}, \tag{3.360}$$

which naturally produced the famous $1/2$ as a vacuum energy. You did not introduce a normal-ordering constant to be later fixed by consistency. You may wonder why we didn't do a similar reasoning for the oscillators of the string. A posteriori, the reason is that not all strings oscillators are physical and the correct vacuum energy should be the one generated by only physical oscillators. In the lightcone we are precisely handling only physical oscillators and therefore it should be possible to obtain the vacuum energy by normal ordering the classical Hamiltonian (which is $L_0$). Doing this literally we find

$$L_0 = \frac{1}{2}(\alpha_0)^2 + \frac{1}{2}\sum_{k\neq 0} \alpha^i_{-k}\alpha^i_k$$
$$= \frac{1}{2}(\alpha_0)^2 + \sum_{k\geq 1} \alpha^i_{-k}\alpha^i_k + \frac{1}{2}\sum_{k\geq 1}\left[\alpha^i_{-k}, \alpha^i_k\right]$$
$$= \frac{1}{2}(\alpha_0)^2 + \sum_{k\geq 1}\alpha^i_{-k}\alpha^i_k + \frac{1}{2}(D-2)\sum_{k\geq 1}k = L_0^{\text{quantum}} - a. \tag{3.361}$$

In doing this we find a divergent normal ordering constant

$$a = -\frac{1}{2}(D-2)\sum_{k\geq 1}k = \infty. \tag{3.362}$$

This is expected because (for each transverse direction $i = 1, \cdots, D-2$) we have infinite oscillators, each one contributing $-\frac{k}{2}$ to $a$. Luckily this divergence can be easily *regularized* and *renormalized* by standard QFT methods in two dimensions. First of all we regulate the divergence in the Hamiltonian by introducing a regulator $\epsilon \to 0$ in the following way:

$$\sum_{k=1}^{\infty}k \longrightarrow \sum_{k=1}^{\infty}k e^{-\epsilon k} = -\frac{1}{2}\frac{1}{1-\cosh\epsilon} = \frac{1}{\epsilon^2} - \frac{1}{12} + \frac{\epsilon^2}{240} + \mathcal{O}(\epsilon^4)$$
$$\stackrel{\epsilon \to 0}{=} \frac{1}{\epsilon^2} - \frac{1}{12}. \tag{3.363}$$

We can then implement the renormalization inserting inside the lightcone action a constant local counterterm $\frac{r}{\epsilon^2}$, a cosmological constant

$$S = \frac{1}{2\pi\alpha'}\int_{\text{WS}} d^2\sigma\left(\partial_+ X^i \partial_- X_i + \frac{r}{\epsilon^2}\right), \tag{3.364}$$

where the constant $r$ can be chosen in such a way that the new Hamiltonian is now finite for $\epsilon \to 0$. Effectively this means that we drop the divergent $\frac{1}{\epsilon^2}$ contribution from the sum (3.363) and we write

$$\left(\sum_{k=1}^{\infty}k\right)_{\text{renorm}} = -\frac{1}{12}. \tag{3.365}$$

We can obtain the same result observing that, recalling $\zeta(s) = \sum_{k=1}^{\infty} \frac{1}{k^s}$ the Riemann $\zeta$ function, the divergent sum can be interpreted as a non convergent representation of the $\zeta$ function evaluated at $-1$:

$$\sum_{k=1}^{\infty} k \to \zeta(-1) = -\frac{1}{12}. \tag{3.366}$$

Therefore, either as two-dimensional quantum-field theorists or as believers in the truth of analytic continuation (the two attitudes luckily give the same result!), we have got rid of the infinite zero point energy and we have found

$$a^{(\text{renorm})} = -\frac{D-2}{2}\left(-\frac{1}{12}\right) = \frac{D-2}{24}. \tag{3.367}$$

But since we previously fixed $a = 1$, the dimension of the target space is fixed to $D = 26$, which is exactly what we found in BRST quantization. All seems consistent and it's not a coincidence, as we will see shortly.

### 3.4.4 Physical states and the partition function

We have seen that the string gives rise to an enormous proliferation of higher-spin massive states. How many physical states we find at every given mass level? Either in OCQ and in BRST this would require to solve the Virasoro constraints and then to carefully find out the null states (or the exact states) to subtract them. In the lightcone, on the other hand, we only have physical oscillators and we can easily count the degeneracy of each mass level. Let us therefore count the states we get by acting with oscillators on the zero momentum vacuum:

$$\alpha_{-k_1}^{i_1} \dots \alpha_{-k_n}^{i_n} |0\rangle. \tag{3.368}$$

This can be done by considering the partition function

$$\text{Tr}\left\{q^{L_0^{D=24}}\right\} = \left(\sum_s \langle s| q^{L_0^{D=1}} |s\rangle\right)^{24}, \tag{3.369}$$

where $s = \{n_1, n_2, \dots, n_k, \dots\}$ (being $n_k$ the occupation number of the $k$-th oscillator) and

$$\langle s|s'\rangle = \delta_{s,s'}. \tag{3.370}$$

Moreover we have used the trivial fact that each transverse direction contributes equally to the partition function. By explicit computation we have

$$\begin{aligned}
\text{Tr}\left\{q^{L_0^{D=1}}\right\} &= \sum_{n_1,\dots,n_\infty=0}^{\infty} \langle\{n_i\}| q^{L_0} |\{n_i\}\rangle \\
&= \sum_{n_1,\dots,n_\infty=0}^{\infty} q^{n_1+2n_2+\dots+kn_k+\dots} \\
&= \sum_{n_1=0}^{\infty} q^{n_1} \sum_{n_2=0}^{\infty} (q^2)^{n_2} \cdots \sum_{n_k=0}^{\infty} (q^k)^{n_k} \cdots \\
&= \frac{1}{1-q}\frac{1}{1-q^2}\cdots\frac{1}{1-q^k}\cdots \\
&= \prod_{k=1}^{\infty} \frac{1}{1-q^k}, \tag{3.371}
\end{aligned}$$

where $\sum_{n=0}^{\infty} q^{kn}$ is the partition function of an harmonic oscillators having $q = e^{k\beta}$.

This all works in one spatial dimension. If we have $D - 2$ spatial euclidean directions, we get

$$\mathrm{Tr}\left\{q^{L_0}\right\} = \left(\prod_{k=1}^{\infty}\frac{1}{1-q^k}\right)^{D-2} = \mathcal{P}(q), \qquad (3.372)$$

which counts the number of states we have at each level. If we expand it for small $q$ and choose $D = 26$, we find

$$\mathcal{P}(q) \overset{q\to 0}{=} 1 + 24q + 324q^2 + \dots \qquad (3.373)$$

This means:

- at $N = 0$ we have 1 state (d.o.f. of the tachyon),

- at $N = 1$ we have 24 states (d.o.f. of the photon),

- at $N = 2$ we have 324 states (d.o.f. of a massive traceless tensor: the massive little group is $SO(D - 1 = 25)$ and thus we have $\frac{1}{2}25(25 + 1) - 1 = 324$ d.o.f., where the $-1$ comes from the trace removal),

- and so on.

An asymptotic analysis on the growth of the number of states with the level $N$ gives

$$\#_{\mathrm{states}}(N) \sim \frac{1}{\sqrt{2}}N^{-\frac{27}{4}}\exp\left(4\pi\sqrt{N}\right), \qquad (3.374)$$

and keeping only the leading contribution

$$\log \#_{\mathrm{states}} \sim 4\pi\sqrt{N} \sim 4\pi\alpha' E, \qquad (3.375)$$

where we have used that (ignoring the momentum) the energy is given by the mass and therefore by the level

$$E \sim m = \sqrt{\frac{N-1}{\alpha'}} \sim \sqrt{\frac{N}{\alpha'}}. \qquad (3.376)$$

To this huge degeneracy (which as we see only depends on the energy of the states) we can associate a *microcanonical* entropy

$$S_{\mathrm{micro}}(E) = K \log \#_{\mathrm{states}}(E) = K\,4\pi\alpha'\,E, \qquad (3.377)$$

where $K$ is Boltzmann constant. The fact that the entropy is linear with the energy (for high energy) means that the *temperature*

$$\frac{1}{KT} = \frac{1}{K}\frac{\partial S}{\partial E} = 4\pi\alpha', \qquad (3.378)$$

is *constant*, it does not change as we change the energy of the system! This constant temperature is called *Hagedorn* temperature:

$$KT_H \equiv \frac{1}{4\pi\alpha'}. \qquad (3.379)$$

A constant temperature is a feature of a system whose (log of the) number of degrees of freedom is proportional to the energy itself so that an increase in energy brings to an increase of states rather than an increase in temperature.

This behaviour is typical of a phase transition and suggests that at high energies string theory will enter a new phase.[21]

---

[21]In hadronic physics, where Hagedorn temperature was originally defined, this is the de-confinement phase transitions, where QCD flux tubes (the effective strings) disintegrate and quarks and gluons can finally exist as individual particles.

### 3.4.5 Lightcone vs BRST

Both lightcone and BRST quantization methods fixed

$$\begin{cases} a = 1, \\ D = 26. \end{cases} \tag{3.380}$$

To have a glimpse that this is not a coincidence let us write

$$L_0^{\text{tot}} = L_0^{(\text{matter})} + L_0^{(\text{ghost})} = \frac{1}{2}(\alpha_0)^2 + N^{(\text{matter})} + N^{(\text{ghost})}, \tag{3.381}$$

with

$$N^{(\text{matter})} = \frac{1}{2}\sum_{k\neq 0}\alpha_{-k}\cdot\alpha_k = N^{(\pm)} + N^{\perp}, \tag{3.382a}$$

$$N^{(\text{ghost})} = \sum_{k\in\mathbb{Z}}k\,b_{-k}c_k. \tag{3.382b}$$

In BRST quantization, physical states are at $gh = 1$ and are built on the vacuum $|\downarrow\rangle = c_1\,|0\rangle$. Let us then play the previous game and normal order the total Hamiltonian $L_0$ on the $|\downarrow\rangle$ vacuum to obtain the corresponding zero-point energy. Let's then consider the classical level operators for the ghosts and for the lightcone directions:

$$N^{(\text{ghost})} = \sum_{k\in\mathbb{Z}}k\,b_{-k}c_k = \sum_{k\geq 1}(b_{-k}c_k - b_k c_{-k}) = \sum_{k\geq 1}(b_{-k}c_k + c_{-k}b_k) - \sum_{k\geq 1}k, \tag{3.383}$$

$$N^{(\pm)} = \frac{1}{2}\sum_{k\neq 0}\alpha_{-k}^{\bar{\mu}}\alpha_k^{\bar{\nu}}\eta_{\bar{\mu}\bar{\nu}} = \sum_{k\geq 1}\alpha_{-k}^{\bar{\mu}}\alpha_k^{\bar{\nu}}\eta_{\bar{\mu}\bar{\nu}} + \frac{1}{2}\eta_{\bar{\mu}}^{\bar{\mu}}\sum_{k\geq 1}k$$

$$= \sum_{k\geq 1}\alpha_{-k}^{\bar{\mu}}\alpha_k^{\bar{\nu}}\eta_{\bar{\mu}\bar{\nu}} + \sum_{k\geq 1}k. \tag{3.384}$$

We then realize that ghosts' zero point energy cancels out the lightcone zero-point energy:

$$\text{BRST} = \text{matter} + \text{ghosts} = (\cancel{\text{LC}} + \text{transverse}) + \cancel{\text{ghosts}}. \tag{3.385}$$

Then we are only left with the normal-ordering constant for the transverse direction which we discussed in the previous subsection.

Therefore the idea is that the lightcone represents the gauge invariant content of the BRST system of the string. It can be shown (but we will not do it) that in the Polyakov path integral the integration over the ghosts precisely cancels with the integration over the non-zero mode part of the two lightcone directions, leaving us with a path integral over the transverse directions, where there is no more local gauge redundancy (remember that constant worldsheet translations are not fixed in the approach we are considering)

$$\int \cancel{\mathcal{D}b}\,\cancel{\mathcal{D}c}\,\cancel{\mathcal{D}X^{\pm}}\cancel{\mathcal{D}X^{-}}\,\mathcal{D}X^i\,(\cdots)e^{S_{\text{BRST}}[X^{\pm},X^i,b,c]} = \int \mathcal{D}X^i\,dx_0^+\,dx_0^-\,(\cdots)e^{S_{\text{LC}}[x_0^{\pm},X^i]}. \tag{3.386}$$

The price for this is the loss of Lorentz covariance and, to a more technical level, a major difficulty in developing string perturbation theory. We will now restore the full Lorentz invariance and conformal symmetry but will keep in mind the lightcone for future use.

# 4 Conformal field theory on the worldsheet

## 4.1 CFT and string theory

We have seen in the previous chapter that our gauge fixing of the Polyakov path integral relies on the choice of a fiducial metric

$$h_{\alpha\beta} = \hat{h}_{\alpha\beta} \,. \tag{4.1}$$

As we have already anticipated, however, this gauge condition does not fix the gauge group completely. The reason is that we can consider the so-called *conformal diffeomorphisms* whose effect is to change the metric by a scale factor. Infinitesimally this means

$$\delta_{\text{conf-diff}}\hat{h}_{\alpha\beta} = \nabla_\alpha \xi_\beta + \nabla_\beta \xi_\alpha = -2\omega(\sigma)\hat{h}_{\alpha\beta} \,. \tag{4.2}$$

A vector field satisfying (locally) this equation is called a (local) *conformal Killing vector* and the equation it satisfies is called *conformal Killing equation*. This variation of the metric can now be compensated by a Weyl transformation (recall the definition eq. (3.30)) such that

$$\delta_{\text{Weyl}}\hat{h}_{\alpha\beta} = 2\omega(\sigma)\hat{h}_{\alpha\beta} \,, \tag{4.3}$$

so that in total we have

$$\left(\delta_{\text{conf diff}} + \delta_{\text{Weyl}}\right)\hat{h}_{\alpha\beta} = 0 \,. \tag{4.4}$$

This means that there is a residual gauge symmetry that does not change the fiducial metric. Therefore the gauge fixing is not complete. The Virasoro constraints, implemented by the BRST charge, have precisely the role of taking care of these extra gauge symmetries. Understanding in detail the nature and the consequences of these unfixed gauge symmetries is the main subject of this chapter.

To start with, let us give a useful analogy for the particle. In the free particle we fixed $e = 1$ and we ended up with

$$S = \int d\tau \, \frac{1}{2} \left( \dot{X}^2 - m^2 \right) \,. \tag{4.5}$$

This action is clearly no more invariant under reparametrizations, but still has a residual invariance under a subgroup of reparametrizations given by the constant translations

$$\tau \; \longrightarrow \; \tau + \delta\tau \,, \quad \text{with} \quad \delta\tau = \text{const,} \tag{4.6}$$

which is a complicated way of saying that the Lagrangian is $\tau$-independent. If the lagrangian is independent of time, the Hamiltonian $\hat{H}$ is conserved and it generates time translations in the state space. But if time translations are a residual redundancy then physical configurations should not change under such a transformation. Then, for a physical configuration, the Hamiltonian is not just conserved but it is *vanishing*. This is why we have to impose the condition

$$\hat{H} \, |\text{state}\rangle = \left( \hat{P}^2 + m^2 \right) |\text{state}\rangle = 0 \,. \tag{4.7}$$

For the string the analogue of the time translations are the conformal diffeomorphisms and the analogue of the Hamiltonian constraint are the Virasoro constraints which are the generators of the conformal transformations.

If we choose the usual flat fiducial metric in worldsheet lightcone coordinates

$$\hat{h}_{\alpha\beta} = \begin{pmatrix} 0 & -\frac{1}{2} \\ -\frac{1}{2} & 0 \end{pmatrix} \,, \tag{4.8}$$

meaning that $h_{+-} = h_{-+} = -\frac{1}{2}$ and $h_{++} = h_{--} = 0$ the conformal Killing equations read as follows:

$$\partial_+ \xi_+ = \partial_+ \xi^- = 0, \tag{4.9a}$$

$$\partial_- \xi_- = \partial_- \xi^+ = 0, \tag{4.9b}$$

with $\omega \sim \partial_+ \xi^+ + \partial_- \xi^-$. This means that the residual diffeomorphisms are generated by holomorphic vector fields $(\xi^+(\sigma^+), \xi^-(\sigma^-))$.

Let us now consider the corresponding "matter+ghost" string action in worldsheet light-cone coordinates (3.43):

$$S[X, b, c] = \frac{1}{2\pi\alpha'} \int_{\mathrm{WS}} d^2\sigma_\pm \, \partial_+ X^\mu \partial_- X_\mu - \frac{i}{2\pi} \int_{\mathrm{WS}} d^2\sigma_\pm \left( b_{++} \partial_- c^+ + b_{--} \partial_+ c^- \right), \tag{4.10}$$

and Wick-rotate the Minkowskian time coordinate $\tau$ to an Euclidean coordinate $t$:

$$\tau \to -it. \tag{4.11}$$

Accordingly we define

$$\tau + \sigma \to -it + \sigma = -i(t + i\sigma) \equiv -iw, \tag{4.12a}$$

$$\tau - \sigma \to -it - \sigma = -i(t - i\sigma) \equiv -i\bar{w}, \tag{4.12b}$$

where $w = t + i\sigma$ is the Euclidean version of $\sigma^+$, while $\bar{w} = t - i\sigma$ is the euclidean version of $\sigma^-$. The action (4.10) now reads as

$$S[X, b, c] = \frac{1}{2\pi\alpha'} \int_{\mathrm{WS}} d^2w \, \partial X^\mu \bar{\partial} X_\mu + \frac{1}{2\pi} \int_{\mathrm{WS}} d^2w \left( b\bar{\partial} c + \bar{b} \partial \bar{c} \right), \tag{4.13}$$

where $d^2w = dw d\bar{w}$ and we have denoted

$$\partial = \partial_w, \qquad \bar{\partial} = \partial_{\bar{w}}, \tag{4.14a}$$

$$b = b_{ww}, \qquad \bar{b} = b_{\bar{w}\bar{w}}, \tag{4.14b}$$

$$c = c^w, \qquad \bar{c} = c^{\bar{w}}. \tag{4.14c}$$

This action is invariant under

$$w \longrightarrow f(w), \quad \text{such that } \bar{\partial} f = 0, \tag{4.15a}$$

$$\bar{w} \longrightarrow \bar{f}(\bar{w}), \quad \text{such that } \partial \bar{f} = 0. \tag{4.15b}$$

These conditions on $f$ and $\bar{f}$ are precisely the Cauchy-Riemann equations. Therefore the Wick-rotated action $S[X, b, c]$ is invariant under holomorphic change of coordinates, also known as *conformal transformations*. The fields transform as follows.

- The $X$ field has no $w$-indices and hence does not transform under conformal maps:

$$\boxed{X(w) \xrightarrow{\;f\;} X(f(w)).} \tag{4.16}$$

This means that $X$ is a scalar under conformal transformations.

- The $b$ field has two lower $w$-indices and hence, under a conformal map, $f(w) = z$ transforms as a quadratic differential:

$$b_{ww}(dw)^2 = b_{zz}(dz)^2 \implies b(w) = \left(\frac{dz}{dw}\right)^2 b(z).$$  (4.17)

Therefore, if we write $f'(w) = \frac{\partial f(w)}{\partial w} = \frac{\partial z}{\partial w}$, we have that

$$\boxed{b(w) \xrightarrow{f} \left[f'(w)\right]^2 b\left(f(w)\right).}$$  (4.18)

This means that $b$ is a primary field of weight 2.

- The $c$ field has one upper $w$-index and hence under conformal map $f(w) = z$ transforms as a vector field:

$$c^w \partial_w = c^z \partial_z \implies c(w) = \left(\frac{dz}{dw}\right)^{-1} c(z).$$  (4.19)

This means

$$\boxed{c(w) \xrightarrow{f} \left[f'(w)\right]^{-1} c\left(f(w)\right),}$$  (4.20)

so that $c$ is a primary field of weight $-1$.

### 4.1.1 From the cylinder to the complex plane

The coordinates $w = t + i\sigma$ describe an infinite cylinder of circumference $2\pi$, because $\sigma \sim \sigma + 2\pi$. As shown in fig. 4.1 , it can be represented as a portion of the $w$ complex plane. This portion can be mapped to the whole $\mathbb{C}$ through the exponential map

$$z = e^w.$$  (4.21)

We notice that

$$z = 0 \quad \longrightarrow \quad t = -\infty,$$  (4.22a)
$$z = \infty \quad \longrightarrow \quad t = +\infty,$$  (4.22b)
$$|z| = 1 \quad \longrightarrow \quad t = 0.$$  (4.22c)

Moreover equal time surfaces ($t = $ const) are mapped to circles of constant radius in the complex $z$-plane. Therefore, time-translations $t \to t + a$ are mapped to dilatations $z \to ze^a$. For this reason the dilatations generator on the $z$-plane can be seen as the Hamiltonian of the system (which generates time shifts).

There are some special fields which have a simple transformation rule in going from the cylinder to the complex plane:

$$\boxed{\phi_{\mathrm{cyl}}(w,\bar{w}) = \left(\frac{dz}{dw}\right)^h \left(\frac{d\bar{z}}{d\bar{w}}\right)^{\bar{h}} \phi_{\mathbb{C}}(z,\bar{z}).}$$  (4.23)

Such fields are called *primary fields* of conformal weight $(h, \bar{h})$. We have that

- $X$ is a primary field of weight $(0, 0)$.[22]

---

[22]To be precise, $X$ is a primary of weight zero from the point of view of its conformal transformation. However, as we will see, its 2-point function differs from the standard 2-point functions of primary fields (which is always power-law in the distance) and it is in fact logarithmic.

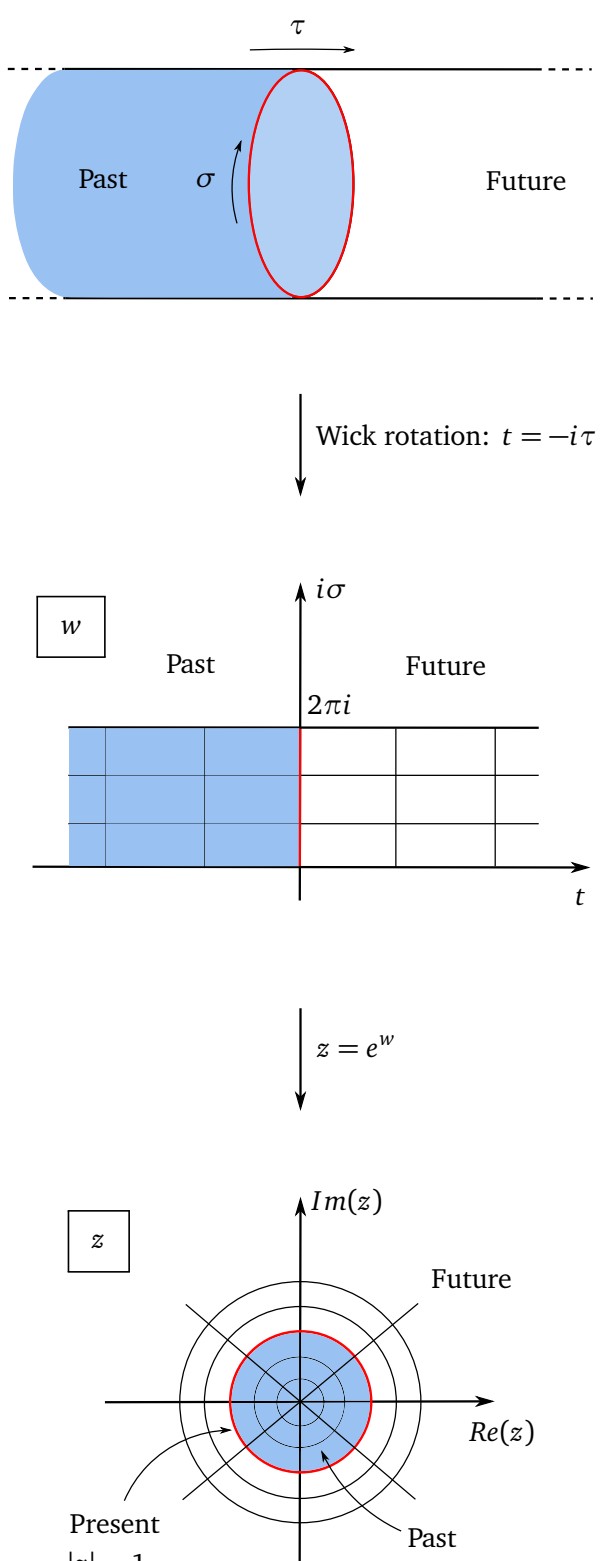

Figure 4.1: The cylinder-like worldsheet of a closed string can be mapped to an infinite strip of height $2\pi i$ on the Euclidean plane of coordinate $w$ after a Wick rotation of the time coordinate. With the exponential function it can then be mapped to the whole complex plane of coordinate $z$, where the ($t = 0$) slice of the WS corresponds to the unitary circle $|z| = 1$ (marked in red): the past ($t < 0$, marked in blue) lies inside such a circle, while the future ($t > 0$) fills the external region.

- $\partial X$ is a primary field of weight $(1,0)$, while $\bar{\partial} X$ is a primary field of weight $(0,1)$.

- $b$ is a primary field of weight $(2,0)$, while $\bar{b}$ is a primary field of weight $(0,2)$.

- $c$ is a primary field of weight $(-1,0)$, while $\bar{c}$ is a primary field of weight $(0,-1)$.

Notice that inside the string action (4.13) we only have terms of weight $(1,1)$.

Assuming for the being that $(h,\bar{h})$ are integer, let us now consider the holomorphic part $\phi^{(h)}(w)$ of a primary field $\phi^{(h,\bar{h})}(w,\bar{w})$ of weight $(h,\bar{h})$ and write (using $z = e^{w}$)

$$\phi_{\text{cyl}}^{(h)}(w) = \sum_{n\in\mathbb{Z}}\phi_n e^{-nw} = \sum_{n\in\mathbb{Z}}\phi_n(z(w))^{-n} = \sum_{n\in\mathbb{Z}}\phi_n z^{-n}, \tag{4.24}$$

$$\phi_{\mathbb{C}}^{(h)}(z) = \left(\frac{dw}{dz}\right)^h \phi_{\text{cyl}}^{(h)}(w) = \left(\frac{d\log z}{dz}\right)^h \phi_{\text{cyl}}^{(h)}(w) = z^{-h}\sum_{n\in\mathbb{Z}}\phi_n z^{-n}. \tag{4.25}$$

Therefore the expansion of an holomorphic primary field of weight $h$ in the complex plane is given by

$$\phi_{\mathbb{C}}^{(h)}(z) = \sum_{n\in\mathbb{Z}}\phi_n z^{-n-h}. \tag{4.26}$$

A way to memorize this is to assign to $z^{-n-h}$ the natural weight $n+h$ and to $\phi_n$ the weight $-n$ so that we obtain that $\phi_{\mathbb{C}}^{(h)}$ has weight $h$. The modes $\phi_n$ can be written as

$$\phi_n = \oint_0 \frac{dz}{2\pi i} z^{n+h-1}\phi^{(h)}(z), \tag{4.27}$$

and they are therefore non-local operators, since they are circular integrals around zero (see fig. 4.2).

> **Exercise 4.1.1**
>
> Use the residue theorem to show that (4.26) is the inverse of (4.27), and viceversa.

The primary fields we will be interested in can be expanded as follows:

$$j^{\mu}(z) \equiv i\sqrt{\frac{2}{\alpha'}}\partial X^{\mu} = \sum_{n\in\mathbb{Z}}\alpha_n^{\mu}z^{-n-1}, \tag{4.28a}$$

$$b(z) = \sum_{n\in\mathbb{Z}}b_n z^{-n-2}, \tag{4.28b}$$

$$c(z) = \sum_{n\in\mathbb{Z}}c_n z^{-n+1}. \tag{4.28c}$$

So we see that, in a sense, these primary fields can be considered as a way to package all the oscillators we have encountered so far, inside holomorphic operators.

Let's have a look at the conformal group nearby the identity and let us thus consider an infinitesimal conformal transformation

$$z'(z) = z + \epsilon(z), \tag{4.29}$$

where $\epsilon(z)$ is an analytic function of $z$ in a region around the origin.

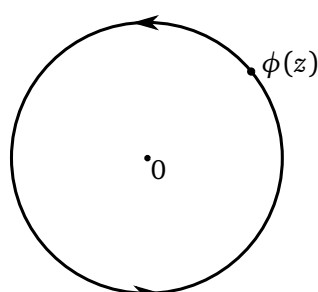

Figure 4.2: The modes $\phi_n$ are integrals of $\phi(z)$ on a closed path around zero.

Under such a map, a primary field $\phi^{(h)}(z)$ is transformed as

$$\phi^{(h)}(z) \longrightarrow \phi^{'(h)}(z') = \left(\frac{dz}{dz'}\right)^h \phi^{(h)}(z). \tag{4.30}$$

In particular, since $\frac{dz}{dz'} = (1 + \partial \epsilon(z))^{-1}$, we have

$$\begin{aligned}
\phi^{'(h)}(z') = \phi^{'(h)}(z + \epsilon(z)) &= (1 + \partial \epsilon(z))^{-h} \phi^{(h)}(z) \\
&= (1 - h \partial \epsilon(z)) \phi^{(h)}(z).
\end{aligned} \tag{4.31}$$

This means that, if we translate $z$ up to $\mathcal{O}(\epsilon^2)$, we get[23]

$$\begin{aligned}
\phi^{'(h)}(z) &= (1 - h \partial \epsilon(z)) \phi(z - \epsilon(z)) \\
&= (1 - h \partial \epsilon(z))(\phi(z) - \epsilon(z) \partial \phi(z)) \\
&= \phi(z) - h \partial \epsilon(z) \phi(z) - \epsilon(z) \partial \phi(z) + \mathcal{O}(\epsilon^2) \\
&= \phi(z) - (\epsilon \partial \phi(z) + h(\partial \epsilon) \phi(z)) + \mathcal{O}(\epsilon^2) \\
&= \phi(z) + \delta_\epsilon \phi(z) + \mathcal{O}(\epsilon^2),
\end{aligned} \tag{4.32}$$

where we defined

$$\boxed{\delta_\epsilon \phi(z) = -(\epsilon \partial \phi(z) + h(\partial \epsilon) \phi(z)).} \tag{4.33}$$

Since $\epsilon(z)$ is naturally the component of the vector field $\epsilon(z)\partial_z$, we can expand it as a $h = -1$ object

$$\boxed{\epsilon(z) = \sum_{n \in \mathbb{Z}} \epsilon_n z^{-n+1}.} \tag{4.34}$$

In the same way we can expand the vector field $\epsilon(z)\partial_z$, namely

$$\boxed{\delta_\epsilon(z) = \epsilon(z)\partial_z = \sum_{n \in \mathbb{Z}} \epsilon_n z^{-n+1} \partial_z = -\sum_{n \in \mathbb{Z}} \epsilon_n l_{-n},} \tag{4.35}$$

where $\{l_n\}$ is a basis of linearly independent vector fields defined as

$$\boxed{l_n = -z^{n+1} \partial_z,} \tag{4.36}$$

and satisfying the algebra

$$\boxed{[l_n, l_m] = (n - m) l_{n+m}.} \tag{4.37}$$

---

[23]Alternatively we can use the universal formula $\phi'(z') = \phi(z) + \delta\phi(z) + \partial\phi(z)\delta z$, when $z' = z + \delta z$.

**Exercise 4.1.2**

Prove eq. (4.37).

Moreover, using the definition (4.35), one can show that

$$\left[\delta_{\epsilon_1}, \delta_{\epsilon_2}\right] = \delta_{\epsilon_1 \overset{\leftrightarrow}{\partial} \epsilon_2}. \tag{4.38}$$

**Exercise 4.1.3**

Prove eq. (4.38).

Given these definitions for an infinitesimal conformal transformation, we can write the finite transformation as

$$f(z) = e^{\epsilon(z)\partial_z} z. \tag{4.39}$$

### 4.1.2 Radial ordering and equal radius commutator

In quantum field theory we use to evaluate correlation functions of time-ordered fields, namely $\langle 0 | T\left[\phi_1(w_1)\phi_2(w_2)\right] | 0 \rangle$. Using the cylinder coordinates $w = t + i\sigma$, the usual definition of the time ordering $T[\dots]$ is

$$T\left[\phi_1(w_1)\phi_2(w_2)\right] = \begin{cases} \phi_1(w_1)\phi_2(w_2), & \text{if } Re(w_1) > Re(w_2), \\ \phi_2(w_2)\phi_1(w_1), & \text{if } Re(w_1) < Re(w_2). \end{cases} \tag{4.40}$$

On the other hand, if we change coordinates with the exponential map (4.21), time ordering turns into *radial ordering*. Indeed, as already said above, equal time surfaces ($t = \text{const}$) are mapped to circles of constant radius (see in fig. 4.1 how slices of the worldsheet are mapped to concentric rings in the $z$ plane). The radial ordering definition is

$$R\left[\phi_1(z_1)\phi_2(z_2)\right] = \begin{cases} \phi_1(z_1)\phi_2(z_2), & \text{if } |z_1| > |z_2|, \\ \phi_2(z_2)\phi_1(z_1), & \text{if } |z_1| < |z_2|. \end{cases} \tag{4.41}$$

If we are dealing with fermions, the radial ordering gains a minus sign, which produces the correct grassmanality.

Given these definitions, the commutator at equal times turns into the commutator at equal radii:

$$\left[\phi_1(z_1), \phi_2(z_2)\right]\Big|_{|z_1|=|z_2|} = \lim_{\varepsilon \to 0^+} \left\{ \phi_1(z_1)\phi_2(z_2)\Big|_{|z_1|=|z_2|+\varepsilon} - \phi_2(z_2)\phi_1(z_1)\Big|_{|z_1|=|z_2|-\varepsilon} \right\}. \tag{4.42}$$

### 4.1.3 Energy-momentum tensor

In any CFT there must be an operator which generates the conformal transformations. This is the energy momentum tensor. We have already encountered it for the CFT of free boson $X^\mu$ and the $b, c$ ghosts. As we have seen in all of those examples, it is a traceless (recall eq. (3.20)) rank two symmetric tensor whose components are

$$\begin{cases} T_{zz} = T(z), \\ T_{\bar{z}\bar{z}} = \bar{T}(\bar{z}). \end{cases} \tag{4.43}$$

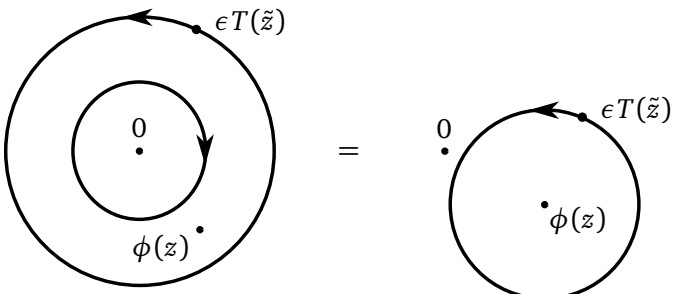

Figure 4.3: The integration paths involved in the evaluation of $\delta_\epsilon \phi(z)$.

Given its geometrical analogy to a symmetric quadratic differential, it makes sense to Taylor-expand it as a $h = 2$ field:

$$T(z) = \sum_{n \in \mathbb{Z}} L_n z^{-n-2} \,. \tag{4.44}$$

We then define the infinite conserved charges

$$T_\epsilon = \oint_0 \frac{dz}{2\pi i} \epsilon(z) T(z) \,. \tag{4.45}$$

Now we insist that the infinitesimal conformal transformation generated by a vector field $\epsilon(z)$ acting on a given field $\phi(z)$ should be obtained through an adjoint action of the generator $T_\epsilon$:

$$\delta_\epsilon \phi(z) = -[T_\epsilon, \phi(z)] \,. \tag{4.46}$$

Recalling now eq. (4.45) and eq. (4.42), we can finally calculate the infinitesimal transformation of a field $\phi(z)$:

$$\begin{aligned}
\delta_\epsilon \phi(z) &= -\left[ \oint_0 \frac{d\tilde{z}}{2\pi i} \epsilon(\tilde{z}) T(\tilde{z}), \phi(z) \right] \\
&= -\left[ \left( \oint_{|\tilde{z}|>|z|} \frac{d\tilde{z}}{2\pi i} - \oint_{|\tilde{z}|<|z|} \frac{d\tilde{z}}{2\pi i} \right) \epsilon(\tilde{z}) T(\tilde{z}) \phi(z) \right] \\
&= -\oint_z \frac{d\tilde{z}}{2\pi i} \epsilon(\tilde{z}) T(\tilde{z}) \phi(z) \,,
\end{aligned} \tag{4.47}$$

where, as shown in fig. 4.3, in the second line we have that $\epsilon(\tilde{z}) T(\tilde{z})$ encircles zero on two paths with two opposite orientation, with the field $\phi(z)$ trapped inside. By contour deformation this is then equivalent to $\epsilon(\tilde{z}) T(\tilde{z})$ circling around $\phi(z)$.

We therefore found that the infinitesimal transformation $\delta_\epsilon$ of a field $\phi(z)$ can be found by integrating $T$ around it, weighed with $\epsilon$. Thus, in order to understand how $\phi(z)$ transforms, we need to know the behaviour of $T$ nearby it, since this is the only contribution to the contour integral. Such an information is contained in the Operator Product Expansion (OPE), which is the topic we will discuss next.

### 4.1.4 Operator product expansion

Let us take two fields $\phi_i(z_i)$ and $\phi_j(z_j)$. Then it is possible to write their product as a series expansion of an infinite number of local fields:

$$\phi_i(z_i) \phi_j(z_j) = \sum_k C_{ij}{}^k (z_i - z_j) \phi_k(z_j) \,, \tag{4.48}$$

where $C_{ij}{}^k(z_1-z_2)$ are the so-called *structure functions*. This series is commonly called *operator product expansion* (OPE).

If we consider fields having a definite weight, their OPE can be further specified as

$$\phi_i^{(h_i)}(z_i)\phi_j^{(h_j)}(z_j) = \sum_k C_{ij}{}^k \phi_k^{(h_k)}(z_j)\frac{1}{(z_i - z_j)^{h_i+h_j-h_k}}\,, \tag{4.49}$$

where $C_{ij}{}^k$ are now called *structure constants*. The OPE can be performed in any QFT, but in a CFT this power series expansion has *infinite* radius of convergence, because a finite radius would result in a scale, which would break conformal invariance.

We now want to determine the OPE of a primary field $\phi^{(h)}(z)$ with the stress-energy tensor $T$. We can parametrize our ignorance by writing

$$T(\tilde{z})\phi^{(h)}(z) = \sum_{k\geq 1}\frac{[T\phi]_k(z)}{(\tilde{z}-z)^k} + \text{regular terms } (k<1)\,, \tag{4.50}$$

where $[T\phi]_k(z)$ are unknown operators. Recalling eq. (4.33), we have that

$$\delta_\epsilon\phi(z) = -\oint_z \frac{d\tilde{z}}{2\pi i}\epsilon(\tilde{z})T(\tilde{z})\phi(z) \stackrel{\text{must}}{=} -(\epsilon\partial\phi(z) + h(\partial\epsilon)\phi(z))\,. \tag{4.51}$$

Using (4.50) and evaluating the residues we can easily see that

- $\epsilon(\partial\phi)$ fixes the single pole:

$$k=1 \implies [T\phi]_1(z) = \partial\phi(z)\,, \tag{4.52}$$

- $h(\partial\epsilon)\phi$ fixes the double pole:

$$k=2 \implies [T\phi]_2(z) = h\phi(z)\,, \tag{4.53}$$

- then we have no higher poles:

$$k>2 \implies [T\phi]_{k>2}(z) = 0\,. \tag{4.54}$$

Therefore the final result is

$$T(\tilde{z})\phi^{(h)}(z) = \frac{h\phi(z)}{(\tilde{z}-z)^2} + \frac{\partial\phi(z)}{(\tilde{z}-z)} + \text{(regular terms)}. \tag{4.55}$$

This OPE can be seen as an alternative definition of a primary field of weight $h$.

We now want to evaluate the OPE $T(z_1)T(z_2)$. We shall recall eqs. (4.38) and (4.46) and then determine the $TT$ OPE in such a way as to enforce

$$\begin{aligned}
[\delta_{\epsilon_1},\delta_{\epsilon_2}]\phi(0) &= \delta_{\epsilon_1}\delta_{\epsilon_2}\phi(0) - \delta_{\epsilon_2}\delta_{\epsilon_1}\phi(0)\\
&= [T_{\epsilon_1},[T_{\epsilon_2},\phi(0)]] - [T_{\epsilon_2},[T_{\epsilon_1},\phi(0)]]\\
&= \oint_0 \frac{dz_2}{2\pi i}\oint_{z_2}\frac{dz_1}{2\pi i}\epsilon_1(z_1)\epsilon_2(z_2)T(z_1)T(z_2)\phi(0) \\
&= \delta_{\epsilon_1\overset{\leftrightarrow}{\partial}\epsilon_2}\phi(0)\\
&= -\oint_0 \frac{dz_2}{2\pi i}(\epsilon_1\partial\epsilon_2(z_2) - \epsilon_2\partial\epsilon_1(z_2))\,T(z_2)\phi(0)
\end{aligned} \tag{4.56}$$

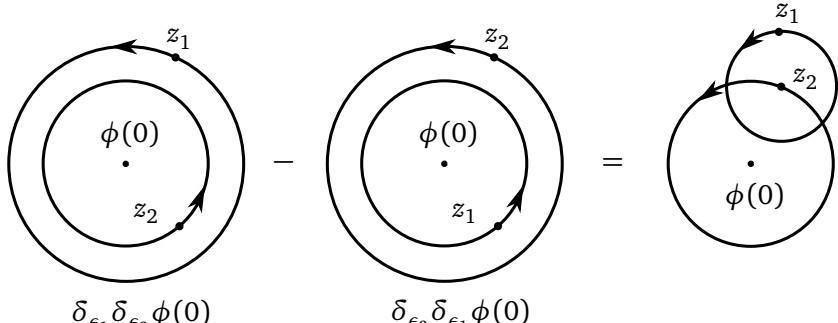

$$\delta_{\epsilon_1}\delta_{\epsilon_2}\phi(0) \qquad\qquad \delta_{\epsilon_2}\delta_{\epsilon_1}\phi(0)$$

Figure 4.4: Representation of the operation $\left(\delta_{\epsilon_1}\delta_{\epsilon_2}-\delta_{\epsilon_2}\delta_{\epsilon_1}\right)\phi(0)$ inside equation (4.56).

(the third step is graphically represented in fig. 4.4).

We can then parametrize our ignorance by writing the generic OPE

$$T(z_1)T(z_2) = \sum_{k\geq 1}\frac{[TT]_k(z_2)}{(z_1-z_2)^k} + \text{regular terms } (k<1)\,, \tag{4.57}$$

that has to reproduce eq. (4.56). Evaluating the contour integral around $z_2$ in the third line of eq. (4.56) we get

$$\sum_{k\geq 1}\oint_0\frac{dz_2}{2\pi i}\epsilon_2(z_2)\frac{\partial^{k-1}\epsilon_1(z_2)}{(k-1)!}[TT]_k(z_2)\phi(0) \stackrel{\text{must}}{=} -\oint_0\frac{dz_2}{2\pi i}\left(\epsilon_1\partial\epsilon_2(z_2)-\epsilon_2\partial\epsilon_1(z_2)\right)T(z_2)\phi(0)\,. \tag{4.58}$$

We then proceed as follows.

- At $k=1$ we must have a weight three operator, which is naturally $\partial T$

$$[TT]_1 = \alpha\partial T\,. \tag{4.59}$$

Now we integrate by part in the contour integral

$$\oint_0\frac{dz_2}{2\pi i}\epsilon_2\epsilon_1\alpha\partial T(z_2)\phi(0) \stackrel{\text{IBP}}{=} -\alpha\oint_0\frac{dz_2}{2\pi i}\partial(\epsilon_2\epsilon_1)T(z_2)\phi(0)\,, \tag{4.60}$$

where there are obviously no boundary terms, since we are integrating on a cycle.

- At $k=2$ we must have a weight two operator and the obvious choice is $T$ itself

$$[TT]_2 = \beta T\,. \tag{4.61}$$

This gives

$$\beta\oint_0\frac{dz_2}{2\pi i}\epsilon_2\partial(\epsilon_1)T(z_2)\phi(0)\,. \tag{4.62}$$

- At $k=3$ we would have a weight 1 operator, but there is no such operator in the game. Therefore we set

$$[TT]_3 = 0\,. \tag{4.63}$$

- At $k=4$ the possible operator should have vanishing weight. It can be the identity operator which, just like $T$, is present in any CFT. It can appear with a generic coefficient that we cannot fix unless we focus on a concrete CFT:

$$[TT]_4 = \text{constant} \equiv \frac{c}{2}\,. \tag{4.64}$$

The contribution of this term is vanishing

$$\frac{c}{2} \oint_0 \frac{dz_2}{2\pi i} \frac{1}{6} \epsilon_2 \partial^3 \epsilon_1(z_2) \phi(0) = 0 \,, \tag{4.65}$$

because $\epsilon(z)$ is holomorphic at the origin. Therefore this quartic pole in the $TT$ OPE is compatible with $[\delta_{\epsilon_1}, \delta_{\epsilon_2}] = \delta_{\epsilon_1 \overset{\leftrightarrow}{\partial} \epsilon_2}$ for any value of the *central charge c*.

After a proper fixing of $\alpha$ and $\beta$, we get the final result, which is

$$T(z_1)T(z_2) = \frac{c}{2} \frac{1}{(z_1 - z_2)^4} + \frac{2T(z_2)}{(z_1 - z_2)^2} + \frac{\partial T(z_2)}{(z_1 - z_2)} + \text{(regular terms)}. \tag{4.66}$$

**Exercise 4.1.4**

Show that the correct fixing produces $\alpha = 1$ and $\beta = 2$, as written in (4.66).

### 4.1.5 Virasoro algebra

As said above (see eq. (4.44)), the stress-energy tensor decomposes in terms of *Virasoro operators* $L_n$ which can be written as

$$L_n = \oint_0 \frac{dz}{2\pi i} z^{n+1} T(z). \tag{4.67}$$

Their algebra is easily obtained by using the $TT$ OPE in the contour integral

$$[L_n, L_m] = \oint_0 \frac{d\tilde{z}}{2\pi i} \oint_{\tilde{z}} \frac{dz}{2\pi i} \tilde{z}^{m+1} z^{n+1} T(\tilde{z}) T(z)$$
$$= (n - m)L_{n+m} + \frac{c}{12} n(n^2 - 1)\delta_{n+m,0} \,, \tag{4.68}$$

which is the Virasoro algebra.

**Exercise 4.1.5**

Prove eq. (4.68).

If we now consider a primary field (with integer weight $h$) $\phi^{(h)}(z) = \sum_{n \in \mathbb{Z}} \phi_n z^{-n-h}$, we can evaluate

$$\left[L_n, \phi^{(h)}(z)\right] = z^n \left(z\partial + (n+1)h\right) \phi^{(n)}(z), \tag{4.69}$$

$$[L_n, \phi_m] = (n(h-1) - m) \phi_{n+m}. \tag{4.70}$$

**Exercise 4.1.6**

Prove eq. (4.69) and eq. (4.70).

Moreover we have that

$$
\begin{aligned}
\delta_\epsilon T(z) &= -[T_\epsilon, T(z)] \\
&= -\frac{c}{12}\partial^3\epsilon(z) - 2(\partial\epsilon)T(z) - \epsilon\partial T(z).
\end{aligned}
\tag{4.71}
$$

Due to the anomalous term $\frac{c}{12}\partial^3\epsilon(z)$, the stress-energy tensor $T$ is not a primary field.

**Exercise 4.1.7**

Prove eq. (4.71).

### 4.1.6 $SL(2,\mathbb{C})$ subgroup

There are infinitesimal transformations $\epsilon_*(z)$ such that $\partial^3\epsilon_*(z) = 0$, namely

$$
\epsilon_*(z) = \alpha + \beta z + \gamma z^2.
\tag{4.72}
$$

These infinitesimal transformation are generated by $L_{-1}, L_0, L_1$, i.e. by $SL(2,\mathbb{C})$:

$$
L_{-1} \longrightarrow -\frac{d}{dz} \qquad \text{(generates translations)},
\tag{4.73a}
$$

$$
L_0 \longrightarrow -z\frac{d}{dz} \qquad \text{(generates dilatations)},
\tag{4.73b}
$$

$$
L_1 \longrightarrow -z^2\frac{d}{dz} \qquad \text{(generates special conformal transformations)}.
\tag{4.73c}
$$

These transformations can be made finite by exponentiating the genereators

$$
e^{-AL_{-1}} : z \longrightarrow z + A,
\tag{4.74a}
$$

$$
e^{-BL_0} : z \longrightarrow e^B z,
\tag{4.74b}
$$

$$
e^{-CL_1} : z \longrightarrow \frac{z}{1+Cz}.
\tag{4.74c}
$$

Their composition generates the generic transformation $f$ of $SL(2,\mathbb{C})$:

$$
f(z) = \frac{Az+B}{Cz+D}.
\tag{4.75}
$$

Moreover it is useful to notice that a special conformal transformation can be obtained by composing an inversion ($\mathcal{I}$) with a translation ($\mathcal{T}$) and then with an inversion again (namely $\mathcal{I} \circ \mathcal{T} \circ \mathcal{I}$):

$$
z \xrightarrow{\mathcal{I}} \frac{1}{z} \xrightarrow{\mathcal{T}} \frac{1}{z} + C \xrightarrow{\mathcal{I}} \frac{1}{\frac{1}{z}+C} = \frac{z}{1+Cz}.
\tag{4.76}
$$

When a field $\phi(z)$ is primary under such a $SL(2,\mathbb{C})$ transformation, it is called *quasi-primary*. Hence $T$ is a quasi-primary field.

### 4.1.7 Hilbert space

The vacuum of our CFT $|0\rangle$ is universally defined in such a way that

$$
\lim_{z\to 0} T(z)|0\rangle = \text{well defined.}
\tag{4.77}
$$

Decomposing the stress-energy tensor in terms of $L_n$ operators we have

$$\lim_{z \to 0} T(z) |0\rangle = \lim_{z \to 0} \sum_{n \in \mathbb{Z}} L_n z^{-n-2} |0\rangle = \text{well defined}. \tag{4.78}$$

In order to be well-defined in $z \to 0$, singular terms should be absent. This statement fixes the annihilating operators as follows:

$$L_n |0\rangle = 0, \quad \forall n \geq -1. \tag{4.79}$$

Since $L_{-n} = L_n^\dagger$, we have that

$$\langle 0 | L_n = 0, \quad \forall n \leq 1. \tag{4.80a}$$

This vacuum $|0\rangle$ is therefore annihilated from the left and from the right by $(L_{-1}, L_0, L_1)$:

$$L_{-1}, L_0, L_1 |0\rangle = 0, \tag{4.81a}$$

$$\langle 0 | L_{-1}, L_0, L_1 = 0. \tag{4.81b}$$

This means that $|0\rangle$ is the $SL(2, \mathbb{C})$-invariant vacuum.

Given the vacuum, the rest of the Hilbert space can be built by acting on $|0\rangle$ with fields placed in $z = 0$, namely $\phi(0)|0\rangle$. If $\phi$ is a primary field $\phi^{(h)}$ of integer weigth $h \in \mathbb{Z}$ we have that the limit

$$\lim_{z \to 0} \phi^{(h)}(z) |0\rangle = \lim_{z \to 0} \sum_{n \in \mathbb{Z}} \phi_n z^{-n-h} |0\rangle, \tag{4.82}$$

must be well defined. This implies

$$\phi_n |0\rangle = 0, \quad \forall n \geq -h+1, \tag{4.83a}$$

$$\langle 0 | \phi_n = 0, \quad \forall n \leq h-1. \tag{4.83b}$$

#### 4.1.7.1 State-operator correspondence

Given a local operator $\phi(z)$ (i.e. a field on the worldsheet), it can be evaluated in $z = 0$ and then applied to the $SL(2, \mathbb{C})$-invariant vacuum $|0\rangle$, thus identifying the state $|\phi\rangle = \phi(0)|0\rangle$. This is commonly called *state-operator correspondence*:

$$\phi(z) \longleftrightarrow |\phi\rangle = \phi(0)|0\rangle. \tag{4.84}$$

If we then consider a primary field $\phi^{(h)}(z)$, we can write its state-operator correspondence as

$$\phi^{(h)}(z) \longleftrightarrow \left| \phi^{(h)} \right\rangle = \phi^{(h)}(0)|0\rangle = \phi_{-h} |0\rangle. \tag{4.85}$$

We can also notice that the $n$-th derivative of a primary field $\phi^{(h)}(z)$ increases its weight of $n$:

$$\phi^{(h)}(0)|0\rangle = \phi_{-h} |0\rangle, \tag{4.86a}$$

$$\partial \phi^{(h)}(0)|0\rangle = \phi_{-h-1} |0\rangle, \tag{4.86b}$$

$$\frac{1}{2} \partial^2 \phi^{(h)}(0)|0\rangle = \phi_{-h-2} |0\rangle, \tag{4.86c}$$

$$\vdots$$

$$\frac{1}{n!} \partial^n \phi^{(h)}(0)|0\rangle = \phi_{-h-n} |0\rangle. \tag{4.86d}$$

Moreover it is useful to remember that

$$L_0 \phi_n |0\rangle = -n \phi_n |0\rangle \,. \tag{4.87}$$

Finally, we can take a field $\phi$ inserted at $z = 0$ and translate it to a generic $z$ through the adjoint action of the linear operator that generates translations (see eq. (4.74a)):

$$\phi(z) = e^{z L_{-1}} \phi(0) e^{-z L_{-1}} \,. \tag{4.88}$$

We therefore get

$$
\begin{aligned}
e^{z L_{-1}} |\phi\rangle = e^{z L_{-1}} \phi(0) |0\rangle &= e^{z L_{-1}} \phi(0) e^{-z L_{-1}} e^{z L_{-1}} |0\rangle \\
&= \phi(z) e^{z L_{-1}} |0\rangle = \phi(z) |0\rangle \,,
\end{aligned} \tag{4.89}
$$

where we have used that the vacuum is annihilated by $L_{-1}$. This gives a concrete way to extract the operator $\phi(z)$ from the state $|\phi\rangle$ and complete the proof of the operator-state correspondence.

### 4.1.7.2 BPZ conjugation

The BPZ conjugation is a map such that BPZ : $|\cdot\rangle \longrightarrow \langle \cdot |$. We are used to the hermitian conjugation, but the BPZ conjugation is slightly different, since it works as follows:

$$\text{BPZ} : \phi^{(h)}(z) \longrightarrow \mathcal{I} \circ \phi^{(h)}(z), \tag{4.90}$$

recalling that $f \circ \phi^{(h)}(z) = \phi'^{(h)}(z) = \left[ f'(z) \right]^h \phi^{(h)}(f(z))$ and where we have introduced the inversion map

$$\mathcal{I}(z) = -\frac{1}{z} \,, \tag{4.91}$$

which maps $0$ to $\infty$ and viceversa. Therefore we have that

$$\text{BPZ}(|\phi\rangle) = \langle 0 | \, \mathcal{I} \circ \phi(0) = \langle 0 | \, \phi\left(-\frac{1}{z}\right) \frac{1}{z^{2h}} \bigg|_{z=0} \,. \tag{4.92}$$

This allows us to take a primary field $\phi^{(h)}$ and write

$$\left| \phi^{(h)} \right\rangle = \phi_{-h} |0\rangle \implies \text{BPZ}\left( \left\langle \phi^{(h)} \right| \right) = \langle 0 | \, \phi_h (-1)^{2h} \,, \tag{4.93}$$

and also

$$\text{BPZ}(\phi_n) = (-1)^{n-h} \phi_{-n} \,. \tag{4.94}$$

> **Exercise 4.1.8**
>
> Prove eq. (4.93) and eq. (4.94).

Given these definitions, we can also introduce the BPZ scalar product:

$$\langle \phi_1, \phi_2 \rangle = \langle \phi_1 | \phi_2 \rangle = \langle \phi_1(\infty) \phi_2(0) \rangle = \langle \mathcal{I} \circ \phi_1(0) \phi_2(0) \rangle \,. \tag{4.95}$$

As a final remark, the simultaneous presence of BPZ and hermitian conjugation defines a reality condition:

$$(|\phi\rangle)^\dagger = \langle \phi | = \text{BPZ}(|\phi\rangle) \iff |\phi\rangle \text{ is real,} \tag{4.96a}$$

$$(|\phi\rangle)^\dagger = -\langle \phi | = -\text{BPZ}(|\phi\rangle) \iff |\phi\rangle \text{ is imaginary.} \tag{4.96b}$$

Some examples are

- $\alpha_{-1} |0\rangle = i \sqrt{\frac{2}{\alpha'}} \partial X(0) |0\rangle$ is real.

- $c_1 |0\rangle$ is real.

- $c_0 |0\rangle$ is imaginary.

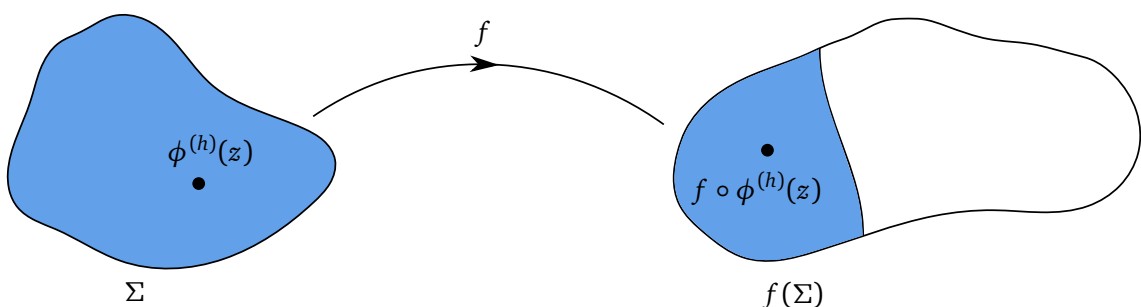

Figure 4.5: Conformal transformation, under the action of the map $f$, of a piece of worldsheet $\Sigma$, where a field $\phi^{(h)}(z)$ is defined.

#### 4.1.7.3  Active conformal transformation

Let us recall the action of a conformal transformation $f(z)$ over operators (graphically represented in fig. 4.5)

$$f \circ \phi^{(h)}(z) = \left[f'(z)\right]^h \phi^{(h)}(f(z)) \,. \tag{4.97}$$

We can write it (thanks to the state-operator correspondence) as a linear transformation acting on the states of the Hilbert space. Thus, let us express the adjoint action of the transformation on the field $\phi(z)$ as

$$f \circ \phi(z) = U_f \phi(z) U_f^{-1} \,, \tag{4.98}$$

where we used the definition

$$U_f = e^{\sum_{n \in \mathbb{Z}} v_n L_{-n}} \,, \tag{4.99}$$

where

$$v(z) = \sum_{n \in \mathbb{Z}} v_n z^{n+1} \,, \tag{4.100}$$

is the vector field (that we previously called $\epsilon(z)$) generating the finite transformation

$$e^{v(z)\partial_z} z = f(z) \,. \tag{4.101}$$

### 4.1.8  Primary states and descendants

A primary state $\phi^{(h)}(z)$ has the properties

$$\boxed{\begin{aligned} L_0 \left|\phi^{(h)}\right\rangle &= h \left|\phi^{(h)}\right\rangle \,, \\ L_{n>0} \left|\phi^{(h)}\right\rangle &= 0 \,. \end{aligned}} \tag{4.102a} \tag{4.102b}$$

For $h = 1$ these are the OCQ Virasoro constraints. We thus see that physical states are primaries of $h = 1$.

> **Exercise 4.1.9**
>
> Prove eq. (4.102a) and eq. (4.102b).

This is like the Cartan decomposition of an algebra: given the Cartan generators (maximal set of mutually commuting generators), we can define primary states that are annihilated by half of the remaining operators (in this case lowering operators). Moreover, the action of

the remaining raising operators generates the *descendant states* from the primary states, thus creating a complete irreducible representation.

In our case we only have one Cartan generator, namely $L_0$, and primary fields are the states with minimal weight in the representation. The lowering (or annihilation) operators are $L_{n>0}$, while the raising (or creation) operators are $L_{n<0}$. In this case the descendants are obtained by acting with Virasoro raising operators on a primary state:

$$L_{-k_1} \cdots L_{-k_{n-1}} L_{-k_n} \left| \phi^{(h)} \right\rangle , \quad k_i > 0 \; \forall i \, . \tag{4.103}$$

We then have that

$$L_0 \left( L_{-k_1} \cdots L_{-k_n} \left| \phi^{(h)} \right\rangle \right) = \left( h + \sum_i k_i \right) \left( L_{-k_1} \cdots L_{-k_n} \left| \phi^{(h)} \right\rangle \right) . \tag{4.104}$$

We thus see that for every primary state we can build an infinite tower of descendants with increasing conformal weights. This infinite tower is called Verma module.

We can rephrase the terminology we have used to classify states in the OCQ, which is now reinterpreted as the CFT of the $X^\mu(z, \bar{z})$ fields.

- A physical state is a primary state of conformal weight 1.

- A spurious state is a descendant of some primary.

- A null state is a descendant that is also a primary. In generic CFT's we are particularly interested in null states of any given possible conformal weight and not just weight=1.

The study of null states is a very important subject in CFT and it opens the door to the so-called *minimal models*. These are CFT's characterized by having only a finite number of primary fields in their spectrum (once the spectrum is quotiented-out by the mull states). They are strongly interacting two-dimensional QFT's that can be fully solved by simply exploiting the power of conformal symmetry. Although this is a very important subject in CFT, it is not needed in this first exposition of string theory and so we direct the interested student to the classical yellow book *Conformal Field Theory* by Di Francesco, Mathieu and Sénéchal.

### 4.1.9 Correlation functions

Let us now consider the $n$-point correlation function

$$\langle \phi_1(z_1) \cdots \phi_n(z_n) \rangle = \langle 0 | R \left[ \phi_1(z_1) \cdots \phi_n(z_n) \right] | 0 \rangle \, , \tag{4.105}$$

which is the same as a Green function of $n$ fields in quantum field theory (from now on the radial ordering $R[\dots]$ will not be explicitly written anymore).

Recalling the fact that, given a $SL(2, \mathbb{C})$ transformation $f$, $|0\rangle$ is $SL(2, \mathbb{C})$-invariant (i.e. $\langle 0 | U_f^{-1} = \langle 0 |$ and $U_f |0\rangle = |0\rangle$), we can easily see that every possible correlation function is invariant under the action of such a conformal transformation $f \in SL(2, \mathbb{C})$:

$$\begin{aligned}
\langle \phi_1(z_1) \cdots \phi_n(z_n) \rangle &= \langle 0 | \phi_1(z_1) \cdots \phi_n(z_n) | 0 \rangle \\
&= \langle 0 | U_f^{-1} U_f \phi_1(z_1) U_f^{-1} U_f \cdots U_f^{-1} U_f \phi_n(z_n) U_f^{-1} U_f | 0 \rangle \\
&= \langle 0 | U_f \phi_1(z_1) U_f^{-1} U_f \cdots U_f^{-1} U_f \phi_n(z_n) U_f^{-1} | 0 \rangle \\
&= \langle 0 | f \circ \phi_1(z_1) \cdots f \circ \phi_n(z_n) | 0 \rangle \, .
\end{aligned} \tag{4.106}$$

We may now focus on the simple cases of 1, 2 and 3-point functions.

#### 4.1.9.1   1-point functions

Let us consider a primary state $\phi^{(h)}(z)$ and evaluate

$$\langle \phi^{(h)}(z) \rangle = \langle \phi^{(h)}(0) \rangle = \langle 0 | \phi^{(h)} \rangle, \tag{4.107}$$

where the first step has been performed thanks to translational invariance of correlation functions (see eq. (4.106)), while the second step is the application of the state-operator correspondence. If we now insert an $L_0$ operator, obtaining $\langle 0 | L_0 | \phi^{(h)} \rangle$, if we let it act on the left it gives 0, while on the right it gives $h \langle 0 | \phi^{(h)} \rangle$. Therefore we can write

$$\boxed{\langle \phi^{(h)}(z) \rangle = A \delta_{h,0},} \tag{4.108}$$

where $A$ is a number. In generic unitary CFT's there won't be dimension zero operators except the identity, however, as we will see, in the ghost conformal field theory (which is not unitary), we will have a weight zero operator with non vanishing one-point function.

#### 4.1.9.2   2-point functions

Let us now consider the 2-point correlation function $\langle \phi_i(z_i) \phi_j(z_j) \rangle$. Let us write the conformal transformation $f$ such that $z_i \to \Lambda \to \infty$ and $z_j \to 0$, namely

$$f(\xi) = \frac{\Lambda}{(z_i - z_j)}(\xi - z_j). \tag{4.109}$$

Therefore we have that

$$
\begin{aligned}
\langle f \circ \phi_i(z_i) f \circ \phi_j(z_j) \rangle &= \left( \frac{\Lambda}{(z_i - z_j)} \right)^{h_i + h_j} \langle \phi_i(\Lambda) \phi_j(0) \rangle \\
&= \left( \frac{\Lambda}{(z_i - z_j)} \right)^{h_i + h_j} \underbrace{\langle \phi_i(\Lambda) \phi_j(0) \rangle \cdot \frac{\Lambda^{-2h_i}}{\Lambda^{-2h_i}}}_{\substack{\text{for } \Lambda \to \infty \text{ it is the} \\ \text{BPZ scalar product}}} \\
&\stackrel{\Lambda \to \infty}{=} \langle \phi_i | \phi_j \rangle \left( \frac{1}{(z_i - z_j)} \right)^{h_i + h_j} \lim_{\Lambda \to \infty} \Lambda^{h_j - h_i}.
\end{aligned}
\tag{4.110}
$$

Therefore, since this correlation function has to be finite, the only possibility is that $h_i = h_j$, which implies that $\lim_{\Lambda \to \infty} \Lambda^{h_j - h_i} = 1$. We can then write the correct result as

$$\boxed{\langle \phi_i(z_i) \phi_j(z_j) \rangle = \frac{G_{ij}}{(z_i - z_j)^{2h}} \delta_{h_i, h_j},} \tag{4.111}$$

where

$$G_{ij} = \langle \phi_i | \phi_j \rangle. \tag{4.112}$$

#### 4.1.9.3   3-point functions

Let us finally consider the 3-point correlation function $\langle \phi_i^{(h_i)}(z_i) \phi_j^{(h_j)}(z_j) \phi_k^{(h_k)}(z_k) \rangle$. Let us write the conformal transformation $f$ such that $z_i \to \infty$, $z_j \to 1$ and $z_k \to 0$, namely

$$f(\xi) = \frac{(z_j - z_i)}{(z_j - z_k)} \frac{(\xi - z_k)}{(\xi - z_i)}, \tag{4.113}$$

which is completely fixed. One can show that

$$\langle \phi_i^{(h_i)}(z_i)\phi_j^{(h_j)}(z_j)\phi_k^{(h_k)}(z_k)\rangle = \frac{C_{ijk}}{(z_i-z_j)^{h_i+h_j-h_k}(z_j-z_k)^{-h_i+h_j+h_k}(z_i-z_k)^{h_i-h_j+h_k}}\,, \tag{4.114}$$

where

$$C_{ijk} = \langle \phi_i| \phi_j(1)|\phi_k\rangle = \langle \mathcal{I}\circ\phi_i(0)\phi_j(1)\phi_k(0)\rangle\,. \tag{4.115}$$

**Exercise 4.1.10**

Prove eq. (4.114).

Higher point correlation functions are also constrained by $SL(2,\mathbb{C})$ but not completely fixed.

## 4.2 Closed string CFT

As we saw in the previous chapter, the worldsheet of a closed string is a cylindrical surface embedded in the targepoint. We can therefore study the physics of the closed string by a proper two-dimensional CFT built on the WS (of coordinate $w$), which can be mapped on the complex plane through the function $z = e^w$ (see fig. 4.1). We can split the matter and the ghost sectors and study the matter $X$-CFT and the ghost $(b,c)$-CFT separately.

### 4.2.1 Matter $X$-CFT

#### 4.2.1.1 Fields and OPEs

The scalar field $X^\mu$ is made of $D$ free bosons living on the worldsheet and its indices $\mu = 0, 1, \ldots, D-1$ are global Lorentz indices. Its action on the complex plane is given by (recall eq. (4.10))

$$S_{\text{matter}} = \frac{1}{2\pi\alpha'}\int_{\text{WS}} d^2z\, \partial X(z,\bar{z})\cdot\bar{\partial}X(z,\bar{z})\,, \tag{4.116}$$

where, as usual, we defined $\partial = \partial_z$ and $\bar{\partial} = \partial_{\bar{z}}$.

Its EOM (being in the closed string case, recall eq. (3.56)) are

$$\partial\bar{\partial}X^\mu = 0\,. \tag{4.117}$$

Therefore in these complex coordinates we can write

$$X^\mu(z,\bar{z}) = X^\mu(z) + \overline{X}^\mu(\bar{z})\,, \tag{4.118}$$

with

$$X^\mu(z) = \frac{X_0^\mu + c^\mu}{2} - i\frac{\alpha'}{2}P^\mu\log z + i\sqrt{\frac{\alpha'}{2}}\sum_{n\neq 0}\frac{1}{n}\alpha_n^\mu z^{-n}\,, \tag{4.119a}$$

$$\overline{X}^\mu(\bar{z}) = \frac{X_0^\mu - c^\mu}{2} - i\frac{\alpha'}{2}P^\mu\log\bar{z} + i\sqrt{\frac{\alpha'}{2}}\sum_{n\neq 0}\frac{1}{n}\alpha_n^\mu\bar{z}^{-n}\,, \tag{4.119b}$$

where, as usual, $X^\mu(z)$ is the holomorphic part of $X^\mu(z,\bar{z})$, while $\overline{X}^\mu(\bar{z})$ is its anti-holomorphic part. In the center of mass momentum term we now have a logarithmic dependence on $z$ because $w = \log z$.

SciPost Phys. Lect. Notes 90 (2025)

Given these fields, we then define the *current* operator (recall eq. (4.28a))

$$j^\mu(z) = i\sqrt{\frac{2}{\alpha'}}\partial X^\mu(z) = \sum_{n\in\mathbb{Z}}\alpha_n^\mu z^{-n-1}\,, \qquad (4.120a)$$

$$\bar{j}^\mu(\bar{z}) = i\sqrt{\frac{2}{\alpha'}}\bar{\partial}\overline{X}^\mu(\bar{z}) = \sum_{n\in\mathbb{Z}}\tilde{\alpha}_n^\mu \bar{z}^{-n-1}\,. \qquad (4.120b)$$

As we will see, it is a primary field of weight 1.

If we focus on its holomorphic part (namely on $j^\mu(z)$) along a specific direction (i.e. omitting the Lorentz index), we can evaluate the 2-point function:

$$\begin{aligned}
\langle j(z_1)j(z_2)\rangle &= \sum_{n,m\in\mathbb{Z}} z_1^{-n-1}z_2^{-m-1}\langle 0|\alpha_n\alpha_m|0\rangle = \sum_{n\geq 0} n z_1^{-n-1}z_2^{n-1}\\
&= \frac{1}{z_1 z_2}\sum_{n\geq 0} n\left(\frac{z_2}{z_1}\right)^n = \frac{1}{z_1 z_2}\frac{\frac{z_2}{z_1}}{\left(1-\frac{z_2}{z_1}\right)^2}\\
&= \frac{1}{(z_1-z_2)^2}\,, \qquad\qquad\qquad\qquad\qquad\qquad (4.121)
\end{aligned}$$

where we chose $|z_1| > |z_2|$ for the radial ordering and recalled that for the closed string $\alpha_0 = \sqrt{\alpha'/2}P$.

If we then consider $D$ free bosons (one for each direction of the target space), we get

$$\langle j^\mu(z_1)j^\nu(z_2)\rangle = \frac{\eta^{\mu\nu}}{(z_1-z_2)^2}\,. \qquad (4.122)$$

Let us now evaluate the OPE $j(z_1)j(z_2)$ using the oscillator algebra. We can consider

$$\alpha_n = \oint_0 \frac{dz}{2\pi i}z^n j(z)\,, \qquad (4.123)$$

and

$$[\alpha_n,\alpha_m] = n\delta_{n+m,0} = \oint_0 \frac{dz_1}{2\pi i}\oint_{z_1}\frac{dz_2}{2\pi i}z_1^n z_2^m j(z_1)j(z_2)\,. \qquad (4.124)$$

We can parametrize the short-distance behaviour as

$$j(z_1)j(z_2) = \frac{A}{(z_1-z_2)^2} + \frac{B}{(z_1-z_2)} + \text{(regular terms)}\,. \qquad (4.125)$$

It can be shown that, in order to find the oscillator algebra (4.124), we have to fix $A = 1$ and $B = 0$. Therefore the final result is

$$j(z_1)j(z_2) = \frac{1}{(z_1-z_2)^2} + \text{(regular terms)}\,. \qquad (4.126)$$

SciPost Phys. Lect. Notes 90 (2025)

**Exercise 4.2.1**

Prove that we have to fix $A = 1$ and $B = 0$ inside (4.125) in order to find the oscillator algebra (4.124).

We shall now introduce a useful notation. Given the OPE $A(z_1)B(z_2)$, we can divide it into its singular part (also called *contraction*), denoted as

$$\underline{A(z_1)B(z_2)}, \tag{4.127}$$

and its regular part, denoted as

$$:A(z_1)B(z_2): \ . \tag{4.128}$$

This $: \cdots :$ is called *normal-ordered product* and, on the $SL(2,\mathbb{C})$-invariant vacuum, it is the same as the oscillator normal ordering we have introduced in previous chapters.

From (4.126) we have that

$$\underline{j(z_1)j(z_2)} = \frac{1}{(z_1 - z_2)^2}\,, \tag{4.129}$$

and therefore

$$\boxed{\begin{aligned} \underline{\partial X(z_1)\partial X(z_2)} &= -\frac{\alpha'}{2}\frac{1}{(z_1 - z_2)^2}\,, & \text{(4.130a)} \\ \underline{\partial X(z_1)X(z_2)} &= -\frac{\alpha'}{2}\frac{1}{(z_1 - z_2)}\,, & \text{(4.130b)} \\ \underline{X(z_1)X(z_2)} &= -\frac{\alpha'}{2}\log(z_1 - z_2)\,. & \text{(4.130c)} \end{aligned}}$$

We observe that the scalar field $X^\mu$ has a logarithmic OPE, which is not what we found for primary fields. This means that $X^\mu$ is not a primary field but, on the other hand, a logarithmic field of weight zero.

The matter stress-energy tensor is given by

$$\boxed{T^{(\text{m})}(z) = -\frac{1}{\alpha'}:\partial X\partial X:(z) = \frac{1}{2}:jj:(z),} \tag{4.131}$$

where we defined

$$\begin{aligned} :AB:(z_1) &= \lim_{z_2 \to z_1}\left(A(z_1)B(z_2) - \underline{A(z_1)B(z_2)}\right) \\ &= \oint_{z_1}\frac{dz_2}{2\pi i}\frac{A(z_1)B(z_2)}{(z_1 - z_2)}\,. \end{aligned} \tag{4.132}$$

We may then evaluate the singular term of the OPE $T^{(\text{m})}(z_1)T^{(\text{m})}(z_2)$ using Wick theorem, which tells us to take all possible contractions and then to normal-order the non-contracted fields; the multiplicity in front of every contraction comes from the fact that some contractions can be taken in multiple ways. The result is the following:

$$\begin{aligned} \underline{T^{(\text{m})}(z_1)T^{(\text{m})}(z_2)} &= \frac{1}{4}\left(\underline{:jj:(z_1)\,:jj:(z_2)}\right) \\ &= \frac{1}{4}\left(2\frac{1}{(z_1 - z_2)^2}\frac{1}{(z_1 - z_2)^2} + 4\frac{1}{(z_1 - z_2)^2}\,:j(z_1)j(z_2):\right) \end{aligned}$$

SciPost Phys. Lect. Notes 90 (2025)

$$= \frac{1}{2} \frac{1}{(z_1 - z_2)^4} + \frac{1}{(z_1 - z_2)^2} : \Big( j(z_2) + (z_1 - z_2) \partial j(z_2) + \dots \Big) j(z_2) :$$

$$= \frac{1}{2} \frac{1}{(z_1 - z_2)^4} + \frac{2 T^{(\mathrm{m})}(z_2)}{(z_1 - z_2)^2} + \frac{:(\partial j) j:(z_2)}{(z_1 - z_2)}$$

$$= \frac{1}{2} \frac{1}{(z_1 - z_2)^4} + \frac{2 T^{(\mathrm{m})}(z_2)}{(z_1 - z_2)^2} + \frac{\partial T^{(\mathrm{m})}(z_2)}{(z_1 - z_2)}, \tag{4.133}$$

where, inside the normal ordering, we expanded $j(z_1)$ around $z = z_2$ using a Taylor expansion up to $\mathcal{O}\big((z_1 - z_2)^2\big)$ (further terms would have given non singular contributions).

Moreover, recalling eq. (4.66), we see that this calculation has fixed the central charge to be

$$\boxed{c = 1.} \tag{4.134}$$

The final OPE is hence

$$\boxed{T^{(\mathrm{m})}(z_1) T^{(\mathrm{m})}(z_2) = \frac{1}{2} \frac{1}{(z_1 - z_2)^4} + \frac{2 T^{(\mathrm{m})}(z_2)}{(z_1 - z_2)^2} + \frac{\partial T^{(\mathrm{m})}(z_2)}{(z_1 - z_2)} + (\text{regular terms}).} \tag{4.135}$$

We can also evaluate

$$T^{(\mathrm{m})}(z_1) j(z_2) = \frac{1}{2} : j j : (z_1) j(z_2)$$

$$= \frac{1}{2} 2 j(z_1) \frac{1}{(z_1 - z_2)^2}$$

$$= \frac{j(z_2) + (z_1 - z_2) \partial j(z_2) + \dots}{(z_1 - z_2)^2}$$

$$= \frac{j(z_2)}{(z_1 - z_2)^2} + \frac{\partial j(z_2)}{(z_1 - z_2)}. \tag{4.136}$$

The final OPE between $T^{(\mathrm{m})}$ and $j$ is hence

$$\boxed{T^{(\mathrm{m})}(z_1) j(z_2) = \frac{j(z_2)}{(z_1 - z_2)^2} + \frac{\partial j(z_2)}{(z_1 - z_2)} + (\text{regular terms}).} \tag{4.137}$$

This means that $j$ is a primary field of weight 1 (recalling eq. (4.55) as a definition of a primary field).

If we now want to create, starting from the $SL(2, \mathbb{C})$-invariant vacuum, a definite-momentum vacuum $|0, P\rangle$, we have to introduce the operator

$$\boxed{\begin{aligned} \mathcal{V}_P(z, \bar{z}) &= :e^{iP \cdot X}:(z, \bar{z}) \\ &= :e^{iP \cdot X_L}:(z) :e^{iP \cdot X_R}:(\bar{z}) \\ &= \mathcal{V}_P(z) \overline{\mathcal{V}}_P(\bar{z}). \end{aligned}} \tag{4.138}$$

It is factorized into its holomorphic and anti-holomorphic component. The normal ordering of a plane wave $:e^{iP \cdot X}:(z)$ has to be understood as the power series

$$:e^{iP \cdot X}:(z) = \sum_{n=0}^{\infty} \frac{(iP)^n}{n!} :X^n:(z), \tag{4.139}$$

where

$$:X^n:(z) = X^n - \overbrace{XX\cdots X}^{n}. \tag{4.140}$$

We can now evaluate

$$
\begin{aligned}
\underline{j(z_1)X(z_2)} &= i\sqrt{\frac{2}{\alpha'}}\,\underline{\partial X(z_1)X(z_2)} \\
&= i\sqrt{\frac{2}{\alpha'}}\left(-\frac{\alpha'}{2}\right)\frac{1}{z_1-z_2} \\
&= -i\sqrt{\frac{\alpha'}{2}}\frac{1}{z_1-z_2},
\end{aligned} \tag{4.141}
$$

then

$$
\begin{aligned}
\underline{j(z_1):X^n:(z_2)} &= n\,\underline{j(z_1)X(z_2)}:X^{n-1}:(z_2) \\
&= -in\sqrt{\frac{\alpha'}{2}}\frac{1}{z_1-z_2}:X^{n-1}:(z_2),
\end{aligned} \tag{4.142}
$$

and finally

$$
\begin{aligned}
\underline{j(z_1):e^{iP\cdot X}:(z_2)} &= \sum_{n=0}^{\infty}\frac{(iP)^n}{n!}\left(-in\sqrt{\frac{\alpha'}{2}}\frac{1}{z_1-z_2}\right):X^{n-1}:(z_2) \\
&= \frac{P\sqrt{\frac{\alpha'}{2}}}{z_1-z_2}\sum_{n=1}^{\infty}\frac{(iP)^{n-1}}{(n-1)!}:X^{n-1}:(z_2) \\
&= \frac{P\sqrt{\frac{\alpha'}{2}}}{z_1-z_2}:e^{iP\cdot X}:(z_2).
\end{aligned} \tag{4.143}
$$

To summarize

$$j(z_1)X(z_2) = -i\sqrt{\frac{\alpha'}{2}}\frac{1}{z_1-z_2} + \text{(regular terms)}, \tag{4.144a}$$

$$j(z_1):X^n:(z_2) = -in\sqrt{\frac{\alpha'}{2}}\frac{1}{z_1-z_2}:X^{n-1}:(z_2) + \text{(regular terms)}, \tag{4.144b}$$

$$j(z_1):e^{iP\cdot X}:(z_2) = \frac{P\sqrt{\frac{\alpha'}{2}}}{z_1-z_2}:e^{iP\cdot X}:(z_2) + \text{(regular terms)}. \tag{4.144c}$$

Via the operator/state correspondence we can write

$$|0,P\rangle = :e^{iP\cdot X}:(z,\bar{z})\Big|_{z=\bar{z}=0}|0\rangle, \tag{4.145}$$

which is the same as what we did previously

$$|0,P\rangle = e^{iP\cdot\hat{X}_0}|0\rangle, \tag{4.146}$$

since our free boson $X(z,\bar{z})$ is

$$X(z,\bar{z}) = \hat{X}_0 + i\hat{P}\log|z|^2 + \text{ oscillators}. \tag{4.147}$$

Finally we have that

$$T^{(m)}(z_1) :e^{iP \cdot X}: (z_2) = \frac{\frac{\alpha' P^2}{4} :e^{iP \cdot X}: (z_2)}{(z_1 - z_2)^2} + \frac{\partial :e^{iP \cdot X}: (z_2)}{(z_1 - z_2)} + \text{(regular terms)}.$$

(4.148)

Therefore the plane wave $:e^{iP \cdot X}: (z)$ is a primary field of weight $\frac{\alpha' P^2}{4}$.

> **Exercise 4.2.2**
>
> Prove eq. (4.148).

### 4.2.1.2  Vertex operators

Through the state-operator correspondence, we define a *vertex operator* (VO) as the wolrdsheet field operator associated to a physical state. For closed strings, a vertex operator is a primary field of weight $(1, 1)$. Let's now proceed to analyze, in this new language of CFT, the first physical states.

■ **Level $N = 0$**

At the level $N = 0$ in the matter sector we have the tachyon, whose vertex operator is given by

$$t(P) :e^{iP \cdot X}: (z, \bar{z}),$$

(4.149)

which has to satisfy the weight one condition

$$h = \frac{\alpha' P^2}{4} = 1,$$

(4.150)

in order to be a weight 1 field. On the other hand, speaking about the primariness, the plane wave is always a primary field and hence no further condition on the momentum $P$ is needed.

■ **Level $N = 1$**

At the level $N = 1$ in the matter sector we have the massless state

$$G_{\mu\nu}(P) \alpha^{\mu}_{-1} \tilde{\alpha}^{\nu}_{-1} |0, P\rangle .$$

(4.151)

The corresponding vertex operator is

$$G_{\mu\nu}(P) :j^{\mu} \bar{j}^{\nu} e^{iP \cdot X}: (z, \bar{z}),$$

(4.152)

its scaling dimension is $(1, 1)$ if

$$\alpha' P^2 = 0,$$

(4.153)

since $j^{\mu}$ and $\bar{j}^{\nu}$ already give the correct $(1, 1)$ weight. However, due to the normal ordering, it is not guaranteed that it is a primary. To ascertain this we compute the OPE with $T(z)$. Restricting to the holomorphic sector, we have that

$$\begin{aligned}
T^{(m)}(z_1) &:j^{\mu} e^{iP \cdot X}: (z, \bar{z}) G_{\mu\nu}(P) \\
&= \frac{\sqrt{\frac{\alpha'}{2}} P^{\mu} G_{\mu\nu}}{(z_1 - z_2)^3} :e^{iP \cdot X}: (z_2) \\
&\quad + \left( \frac{\frac{\alpha' P^2}{4} + 1}{(z_1 - z_2)^2} + \frac{\partial_{z_2}}{(z_1 - z_2)} \right) :j^{\mu} e^{iP \cdot X}: (z_2) G_{\mu\nu} \\
&\quad + \text{(regular terms)}.
\end{aligned}$$

(4.154)

So we get a primary if the coefficient of the cubic pole vanishes. Altogether we recover the physicality conditiions

$$
\begin{cases}
\frac{\alpha' P^2}{4} + 1 = 1 \iff P^2 = 0, \\
P^\mu G_{\mu\nu} = 0.
\end{cases}
\tag{4.155}
$$

If we look at the anti-holomorphic part, we get the additional physical condition

$$
G_{\mu\nu} P^\nu = 0.
\tag{4.156}
$$

These are the same physical conditions we found in (3.128) and (3.133).

> **Exercise 4.2.3**
>
> Prove eq. (4.154).

If we think a little about it (comparing what we found now with (3.128)), we realize that the first term inside (4.154) corresponds to the $L_1$ constraint, as it is expected from our general discussion.

### 4.2.1.3 Correlation functions

Let us now consider correlation functions of currents. By Wick theorem we easily get that

$$
\langle j^\mu(z_1) \rangle = 0,
\tag{4.157a}
$$

$$
\langle j^\mu(z_1) j^\nu(z_2) \rangle = \frac{\eta^{\mu\nu}}{(z_1 - z_2)^2},
\tag{4.157b}
$$

$$
\langle j^\mu(z_1) j^\nu(z_2) j^\rho(z_3) \rangle = 0,
\tag{4.157c}
$$

$$
\langle j^\mu(z_1) j^\nu(z_2) j^\rho(z_3) j^\sigma(z_4) \rangle = \frac{\eta^{\mu\nu}\eta^{\rho\sigma}}{(z_1 - z_2)^2(z_3 - z_4)^2}
$$
$$
+ \frac{\eta^{\mu\rho}\eta^{\nu\sigma}}{(z_1 - z_3)^2(z_2 - z_4)^2}
$$
$$
+ \frac{\eta^{\mu\sigma}\eta^{\nu\rho}}{(z_1 - z_4)^2(z_2 - z_3)^2}.
\tag{4.157d}
$$

The 1-point function vanishes because $h(j) = 1$ (recall eq. (4.108)). The 3-point function vanishes because, whichever contraction we try to take, we are always left with a vanishing 1-point function. In general the correlation function of an odd number of $j$'s is always vanishing.

Let us now consider correlation functions of plane waves. Since we are allowed to split the VO into its holomorphic and anti-holomorphic parts, the correlation function factorizes

$$
\langle \mathcal{V}_{P_1}(z_1, \bar{z}_1) \cdots \mathcal{V}_{P_n}(z_n, \bar{z}_n) \rangle = \langle \mathcal{V}_{P_1}(z_1) \cdots \mathcal{V}_{P_n}(z_n) \rangle \cdot \langle \overline{\mathcal{V}}_{P_1}(\bar{z}_1) \cdots \overline{\mathcal{V}}_{P_n}(\bar{z}_n) \rangle,
\tag{4.158}
$$

we can focus on the holomorphic sector and write

$$
\langle \mathcal{V}_{P_1}(z_1) \cdots \mathcal{V}_{P_n}(z_n) \rangle = \langle \prod_{k=1}^{n} :e^{iP_k \cdot X(z_k)}: \rangle
$$
$$
= \prod_{k<l} (z_k - z_l)^{\frac{\alpha'}{2} P_k \cdot P_l} \cdot \delta\left( \sum_{k=1}^{n} P_k \right).
\tag{4.159}
$$

This holds because

$$
\begin{aligned}
\underbrace{:e^{iP_k \cdot X}:(z_k) :e^{iP_l \cdot X}:(z_l)} &= e^{\overbrace{iP_k \cdot X(z_k)iP_l \cdot X(z_l)}} :e^{iP_k \cdot X}(z_k)e^{iP_l \cdot X}(z_l): \\
&= e^{\frac{\alpha'}{2}P_k \cdot P_l \log(z_k-z_l)} :e^{iP_k \cdot X}(z_k)e^{iP_l \cdot X}(z_l): \\
&= (z_k - z_l)^{\frac{\alpha'}{2}P_k \cdot P_l} :e^{iP_k \cdot X}(z_k)e^{iP_l \cdot X}(z_l): .
\end{aligned}
\tag{4.160}
$$

The Dirac delta conserving the total momentum inside (4.159) comes from the fact that, given the conserved charge $j_0^\mu$ inside $j^\mu(z)$, namely

$$
j_0^\mu = \oint_0 \frac{dz}{2\pi i} j^\mu(z) = \alpha_0^\mu = \sqrt{\frac{2}{\alpha'}} P^\mu ,
\tag{4.161}
$$

we can insert it inside the correlation function (4.159). Since it annihilates the vacuum, we have that

$$
\begin{aligned}
\langle j_0(\mathcal{V}_{P_1}(z_1) \cdots \mathcal{V}_{P_n}(z_n)) \rangle &= \langle 0| j_0(\mathcal{V}_{P_1}(z_1) \cdots \mathcal{V}_{P_n}(z_n)) |0\rangle = 0 \\
&= \langle \oint_{z_1 \cdots z_n} \frac{dz}{2\pi i} j^\mu(z)(\mathcal{V}_{P_1}(z_1) \cdots \mathcal{V}_{P_n}(z_n)) \rangle \\
&= \left(\sum_{k=1}^n P_k\right)\left(\sqrt{\frac{\alpha'}{2}}\right)^n \langle \mathcal{V}_{P_1}(z_1) \cdots \mathcal{V}_{P_n}(z_n) \rangle ,
\end{aligned}
\tag{4.162}
$$

where we used

$$
\begin{aligned}
j_0^\mu \mathcal{V}_{P_k}(z_k) &= \oint_{z_k} \frac{dz}{2\pi i} j^\mu(z)e^{iP_k \cdot X}(z_k) \\
&= \oint_{z_k} \frac{dz}{2\pi i} \underbrace{j^\mu(z)e^{iP_k \cdot X}(z_k)} \\
&= \oint_{z_k} \frac{dz}{2\pi i} \sqrt{\frac{\alpha'}{2}} \frac{P^\mu}{(z-z_k)} \mathcal{V}_{P_k}(z_k) \\
&= \sqrt{\frac{\alpha'}{2}} P^\mu \mathcal{V}_{P_k}(z_k) .
\end{aligned}
\tag{4.163}
$$

Therefore eq. (4.162) implies that

$$
\sum_{k=1}^n P_k = 0 ,
\tag{4.164}
$$

namely the total momentum is conserved inside the $n$-point correlation function.

### 4.2.2 Ghost $(b, c)$-CFT

#### 4.2.2.1 Fields and OPEs

The ghost action in the complex plane is given by (recall eq. (4.10))

$$
S_{\text{ghost}} = \frac{1}{2\pi} \int_{\text{WS}} d^2z \left(b\bar{\partial}c + \bar{b}\partial\bar{c}\right) .
\tag{4.165}
$$

As we said before (see eq. (4.28b)), the $b$ ghost has weight 2 (holomorphic quadratic differential) and can be written as

$$
\begin{cases}
b(z) = \sum_{n\in\mathbb{Z}} b_n z^{-n-2} , \\
b_n = \oint_0 \frac{dz}{2\pi i} z^{n+1} b(z) ,
\end{cases}
\tag{4.166}
$$

while the $c$ ghost has weight $-1$ (holomorphic vector field) and can be written as (recall eq. (4.28c))

$$\begin{cases} c(z) = \sum_{n \in \mathbb{Z}} c_n z^{-n+1}, \\ c_n = \oint_0 \frac{dz}{2\pi i} z^{n-2} c(z). \end{cases} \tag{4.167}$$

The ghosts' modes algebra reads as

$$\boxed{[b_n, c_m] = \delta_{n+m,0},} \tag{4.168}$$

and the corresponding OPE is

$$\boxed{b(z_1)c(z_2) = \frac{1}{(z_1 - z_2)} + \text{(regular terms)}.} \tag{4.169}$$

The ghost stress-energy tensor is defined as

$$\boxed{T^{(\text{gh})}(z) = -\Big(2 :b(\partial c): + :(\partial b)c: \Big)(z),} \tag{4.170}$$

and satisfies

$$\boxed{\begin{aligned} T^{(\text{gh})}(z_1)b(z_2) &= \frac{2b(z_2)}{(z_1 - z_2)^2} + \frac{\partial b(z_2)}{(z_1 - z_2)} + \text{(regular terms)}, & (4.171a) \\ T^{(\text{gh})}(z_1)c(z_2) &= -\frac{c(z_2)}{(z_1 - z_2)^2} + \frac{\partial c(z_2)}{(z_1 - z_2)} + \text{(regular terms)}. & (4.171b) \end{aligned}}$$

> **Exercise 4.2.4**
>
> Prove eq. (4.171a) and eq. (4.171b).

Moreover we have that

$$\underline{b(z_1)b(z_2)} = 0, \tag{4.172a}$$

$$\underline{c(z_1)c(z_2)} = 0. \tag{4.172b}$$

All in all we have that

$$\boxed{T^{(\text{gh})}(z_1)T^{(\text{gh})}(z_2) = \frac{-\frac{26}{2}}{(z_1 - z_2)^4} + \frac{2T^{(\text{gh})}(z_2)}{(z_1 - z_2)^2} + \frac{\partial T^{(\text{gh})}(z_2)}{(z_1 - z_2)} + \text{(regular terms)}.} \tag{4.173}$$

> **Exercise 4.2.5**
>
> Prove eq. (4.173). (Pay attention to the fact that $b$ and $c$ are Grassmann odd!)

We then define the ghost current as

$$\boxed{j^{(\text{gh})}(z) = - :bc: (z) = \sum_{n \in \mathbb{Z}} j_n^{(\text{gh})} z^{-n-1},} \tag{4.174}$$

which obviously has weight $h(j^{(gh)}) = 1$. Although its scaling dimension is 1, it is not a primary

$$T^{(\text{gh})}(z_1)j^{(\text{gh})}(z_2) = \frac{-3}{(z_1-z_2)^3} + \frac{j^{(\text{gh})}(z_2)}{(z_1-z_2)^2} + \frac{\partial j^{(\text{gh})}(z_2)}{(z_1-z_2)} + (\text{regular terms}). \qquad (4.175)$$

Finally, we define the ghost number operator as

$$N_{(\text{gh})} = \oint_0 \frac{dz}{2\pi i} j^{(\text{gh})}(z), \qquad (4.176)$$

which is such that

$$\left[N_{(\text{gh})}, c(z)\right] = c(z), \qquad (4.177a)$$

$$\left[N_{(\text{gh})}, b(z)\right] = -b(z). \qquad (4.177b)$$

Hence $b$ has ghost number $gh = -1$ and $c$ has ghost number $gh = 1$.

#### 4.2.2.2  Ghosts primaries

If we evaluate the OPE with $T^{(\text{gh})}$ of the states

$$c(z), \qquad b(z), \qquad (4.178a)$$

$$c(\partial c)(z), \qquad b(\partial b)(z), \qquad (4.178b)$$

$$\frac{1}{2}c(\partial c)(\partial^2 c)(z), \qquad \frac{1}{2}b(\partial b)(\partial^2 b)(z), \qquad (4.178c)$$

we realize that they are all primary fields. In general in a CFT the $\partial$ of a primary is not a primary because the $\partial$ converts the double pole into a triple pole in the OPE with $T$. But in the ghost-CFT one can see that the terms producing the non-primary structure vanish since they are multiplied by $b^2, (\partial b)^2, \ldots$ (or, analogously, for $c^2, (\partial c)^2, \ldots$), which are vanishing because of their grassmanality.

If we now recall that we have four $c$-ghost vacua, we can associate them with primary fields:

$$|0\rangle \quad \longleftrightarrow \quad \mathbb{1} \qquad\qquad (h = 0), \qquad (4.179a)$$

$$c_1|0\rangle \quad \longleftrightarrow \quad c(z) \qquad\qquad (h = -1), \qquad (4.179b)$$

$$c_0 c_1|0\rangle \quad \longleftrightarrow \quad (\partial c)c(z) \qquad\qquad (h = -1), \qquad (4.179c)$$

$$c_{-1}c_0 c_1|0\rangle \quad \longleftrightarrow \quad \frac{1}{2}(\partial^2 c)(\partial c)c(z) \qquad\qquad (h = 0). \qquad (4.179d)$$

We have

$$\langle 0|\hat{0}\rangle = \langle 0|c_{-1}c_0 c_1|0\rangle = 1 \quad \longleftrightarrow \quad \frac{1}{2}\langle c(\partial c)(\partial^2 c)(z)\rangle = 1. \qquad (4.180)$$

We hence found a non-vanishing 1-point function of a vanishing weight field which is not the identity.

#### 4.2.2.3 Three-ghost correlation function

Given what we said above, we have for a three ghost correlation function the only contribution of a $c$-ghost comes from the first terms on its modes expansion:

$$c(z) = c_{-1}z^2 + c_0 z + c_1. \tag{4.181}$$

One can hence evaluate

$$\langle c(z_1)c(z_2)c(z_3)\rangle = -\det \begin{pmatrix} 1 & 1 & 1 \\ z_1 & z_2 & z_3 \\ z_1^2 & z_2^2 & z_3^2 \end{pmatrix} = -(z_1 - z_2)(z_2 - z_3)(z_1 - z_3). \tag{4.182}$$

> **Exercise 4.2.6**
>
> Prove eq. (4.182).

### 4.2.3 BRST and CFT

Let us first recall from BRST quantization the definition of the BRST charge

$$
\begin{aligned}
Q_B &= \sum_{n\in\mathbb{Z}} c_{-n} L_n^{(\mathrm{m})} + \frac{1}{2} :c_{-n} L_n^{(\mathrm{gh})}: \\
&= \oint_0 \frac{dz}{2\pi i} \left( c\, T^{(\mathrm{m})}(z) + \frac{1}{2} :c\, T^{(\mathrm{gh})}:(z) \right) = \oint_0 \frac{dz}{2\pi i} \widetilde{j}_B(z).
\end{aligned} \tag{4.183}
$$

We can think to the matter sector (m) as a generic CFT of central charge $c = 26$, which cancels the ghost sector (gh) central charge. Moreover, we wrote $\widetilde{j}_B(z)$ because the true current is

$$j_B(z) = c\, T^{(\mathrm{m})}(z) + \frac{1}{2} :c\, T^{(\mathrm{gh})}:(z) + \frac{3}{2} \partial^2 c(z), \tag{4.184}$$

which is a weight 1 primary field, with respect to the total stress-energy tensor

$$T^{(\mathrm{tot})}(z) = T^{(\mathrm{m})}(z) + T^{(\mathrm{gh})}(z). \tag{4.185}$$

The additional term $\frac{3}{2}\partial^2 c(z)$ is needed for the primariness of $j_B(z)$, but in the definition of $Q_B$ it is irrelevant, being a total derivative which gives no contribution to the integral.

Given these definitions, one can show that

$$[Q_B, X^\mu(z)] = \oint_z \frac{d\tilde{z}}{2\pi i} j_B(\tilde{z}) X^\mu(z) = c\partial X^\mu(z), \tag{4.186}$$

and also

$$[Q_B, c(z)] = c\partial c(z), \tag{4.187}$$

that, thanks to the state-operator correspondence, is equivalent to

$$Q_B c_1 |0\rangle = c_1 c_0 |0\rangle. \tag{4.188}$$

Moreover one can show that

$$Q_B^2 = 0, \tag{4.189}$$

whenever $c^{\mathrm{m}} = -c^{\mathrm{gh}} = 26$, namely when the matter central charge cancels the ghost central charge.

We also have that

$$\left[Q_B, j_{\mathrm{gh}}(z)\right] = -j_B(z), \tag{4.190}$$

which, thanks to the state-operator correspondence, is equivalent to

$$\left[N_{\mathrm{gh}}, Q_B\right] = Q_B. \tag{4.191}$$

Therefore $j_B$ generates $Q_B$ as a zero mode, but $j_B$ is also $Q_B$-exact. Its primitive is the ghost current $j_{\mathrm{gh}}$. Notice that while $j_{\mathrm{gh}}$ is non primary, $j_B$ is primary.

**Exercise 4.2.7**

Prove eq. (4.186).

**Exercise 4.2.8**

Prove eq. (4.187).

**Exercise 4.2.9**

Prove eq. (4.189). Use the fact that

$$Q_B^2 = \oint_0 \frac{d\tilde{z}}{2\pi i} \oint_{\tilde{z}} \frac{dz}{2\pi i} \, \underline{j_B(z)j_B(\tilde{z})}, \tag{4.192}$$

and that

$$j_B(z)j_B(\tilde{z}) = \ldots + \frac{c^{(\mathrm{m})} - 26}{12} \frac{1}{(z-\tilde{z})} \left(\partial^3 c \cdot c\right)(\tilde{z}) + \ldots \tag{4.193}$$

**Exercise 4.2.10**

Prove eq. (4.190).

We are now ready to analyze the cohomology. Using the state-operator correspondence we can write

$$
\begin{array}{rcll}
|0\rangle & \longleftrightarrow & \mathbb{1} & (gh = 0), \\[6pt]
c_1 \mathcal{V}_{h=1}^{(\mathrm{m})} |0\rangle & \longleftrightarrow & c\mathcal{V}_{h=1}^{(\mathrm{m})}(z) & (gh = 1), \\[6pt]
c_0 c_1 \mathcal{V}_{h=1}^{(\mathrm{m})} |0\rangle & \longleftrightarrow & c(\partial c)\mathcal{V}_{h=1}^{(\mathrm{m})}(z) & (gh = 2), \\[6pt]
c_{-1} c_0 c_1 |0\rangle & \longleftrightarrow & \frac{1}{2} c(\partial c)(\partial^2 c)(z) & (gh = 3),
\end{array}
$$

(4.194a)
(4.194b)
(4.194c)
(4.194d)

where $\mathcal{V}_{h=1}^{(\mathrm{m})}$ is a physical matter operator of weight $h = 1$.

One can also show that

$$\boxed{\left[Q_B, \mathcal{V}_{h=1}^{(\mathrm{m})}(z)\right] = \partial \left(c\mathcal{V}_{h=1}^{(\mathrm{m})}\right)(z).} \tag{4.195}$$

> **Exercise 4.2.11**
>
> Prove eq. (4.195).

This tells us that there are two ways for a state to be killed by $Q_B$:

1. a local operator $c\mathcal{V}^{(m)}_{h=1}(z)$,

2. a non-local *integrated* operator $\int dz\, \mathcal{V}^{(m)}_{h=1}(z) = \mathcal{V}^{(m)}$.

Then the physical condition (i.e. the BRST charge annihilates the physical state) can be expressed in two possible ways:

1. $\left[Q_B, c\mathcal{V}^{(m)}_{h=1}(z)\right] = 0$,

2. $\left[Q_B, \mathcal{V}^{(m)}\right] = \int dz\, \partial\left(c\mathcal{V}^{(m)}_{h=1}\right)(z) \overset{\text{Stokes}}{=} 0$     (within boundary terms).

We now want to write the complete closed string vertex operator, taking both the holomorphic and the anti-holomorphic part and also making explicit the momentum operator

1. The non-integrated VO is

$$c\bar{c}\mathcal{V}(z,\bar{z}) = \int d^d P\, \Phi_{ij}(P) :c\mathcal{V}_i \bar{c}\overline{\mathcal{V}}_j e^{iP\cdot X}:(z,\bar{z}),$$

(4.196)

where $\Phi_{ij}(P)$ is the spacetime polarization and $\mathcal{V}_i$ is a chiral field of weight $h_i$ (and similarly for $\bar{\mathcal{V}}_j$). The polarization $\Phi_{ij}(P)$ has to satisfy appropriate conditions to make $:\mathcal{V}_i e^{iP\cdot X}:$ a primary. It has $gh = (1,1)$ and is an overall primary of weight $h = (0,0)$ if

$$\alpha' P^2/4 + h_i - 1 = \alpha' P^2/4 + h_j - 1 = 0.$$

(4.197)

2. The integrated VO is instead

$$\int dz d\bar{z}\, \mathcal{V}(z,\bar{z}) = \int dz d\bar{z} \int d^d P\, \Phi_{ij}(P) :\mathcal{V}_i \overline{\mathcal{V}}_j e^{iP\cdot X}:(z,\bar{z}).$$

(4.198)

The worldsheet integration $\int dz d\bar{z}$ effectively lowers the total weight to $h = (0,0)$ and therefore we again have a conformal invariant operator, but this time a non-local one.

## 4.3 Open string CFT

### 4.3.1 Boundary conformal field theory

So far we considered a theory defined on a cylinder with coordinate $w$ that we mapped to the complex plane through the conformal map $z = e^w$. Such a construction was suitable for closed strings. When we have to deal with open strings we have some differences: the worldsheet is a now strip and under the conformal transformation $z = e^w$ it is mapped to the upper part of the complex plane (see fig. 4.6). We will refer to it as the *upper half plane* (UHP).

Therefore, after the transformation, the CFT is defined on a surface having as a boundary the real axis, where $z = \bar{z}$. This is an example of a *boundary conformal field theory*, (BCFT).

The presence of a boundary breaks the full conformal group to the subgroup that leaves the boundary invariant. This means for a given $f$ such that

$$z \longrightarrow f(z), \tag{4.199a}$$

$$\bar{z} \longrightarrow \bar{f}(\bar{z}), \tag{4.199b}$$

when we are on the boundary $z = \bar{z}$ we have to require that

$$f(z) = \bar{f}(\bar{z}), \quad \text{for } z = \bar{z}, \tag{4.200}$$

so that $f$ is real

$$f(z) = \sum_n f_n z^n, \quad \text{with} \quad f_n = \bar{f}_n \iff f_n \in \mathbb{R}. \tag{4.201}$$

Conformal transformations are generated by the stress-energy tensor and, on the Riemann sphere, we had that $T(z)$ generates the transformation $f(z)$, while $\overline{T}(\bar{z})$ generates $\bar{f}(\bar{z})$. On the UHP we have $f(z) = \bar{f}(\bar{z})$ for $z = \bar{z}$, and therefore we have to require that

$$\boxed{T(z) = \overline{T}(\bar{z}), \quad \text{for } z = \bar{z}.} \tag{4.202}$$

If we then recall the usual expansion (4.44) of the stress-energy tensor in terms of Virasoro operators, namely

$$T(z) = \sum_{n \in \mathbb{Z}} L_n z^{-n-2}, \tag{4.203a}$$

$$\overline{T}(\bar{z}) = \sum_{n \in \mathbb{Z}} \tilde{L}_n \bar{z}^{-n-2}, \tag{4.203b}$$

we realize that the condition (4.202) implies

$$\boxed{L_n = \tilde{L}_n,} \tag{4.204}$$

which means that we have only one Virasoro algebra.

Just like the Riemann sphere is preserved by the $SL(2,\mathbb{C})$ subgroup of projective transformations, analogously for the UHP we have the group $SL(2,\mathbb{R})$:

$$f(z) = \frac{Az + B}{Cz + D}, \quad \text{with} \quad A, B, C, D \in \mathbb{R}, \quad (AD - BC) \neq 0. \tag{4.205}$$

It is the largest possible group of transformations which sends the UHP to itself in a one-to-one way.

### 4.3.1.1 Bulk fields and doubling trick

Consider a generic field $\phi(z, \bar{z})$ living in the bulk. If we apply to it an infinitesimal conformal transformation (recalling eq. (4.46)), we have that

$$\begin{aligned}
\delta_{(\epsilon, \bar{\epsilon})} \phi(z, \bar{z}) &= -\left( [T_\epsilon, \phi(z, \bar{z})] + \left[ \overline{T}_{\bar{\epsilon}}, \phi(z, \bar{z}) \right] \right) \\
&= -\oint_z \frac{d\tilde{z}}{2\pi i} \epsilon(\tilde{z}) T(\tilde{z}) \phi(z, \bar{z}) - \oint_{\bar{z}} \frac{d\bar{\tilde{z}}}{2\pi i} \epsilon(\bar{\tilde{z}}) \overline{T}(\bar{\tilde{z}}) \phi(z, \bar{z}) \\
&= -\oint_z \frac{d\tilde{z}}{2\pi i} \epsilon(\tilde{z}) \underline{T(\tilde{z}) \phi(z, \bar{z})} - \oint_{\bar{z}} \frac{d\bar{\tilde{z}}}{2\pi i} \epsilon(\bar{\tilde{z}}) \underline{\overline{T}(\bar{\tilde{z}}) \phi(z, \bar{z})},
\end{aligned} \tag{4.206}$$

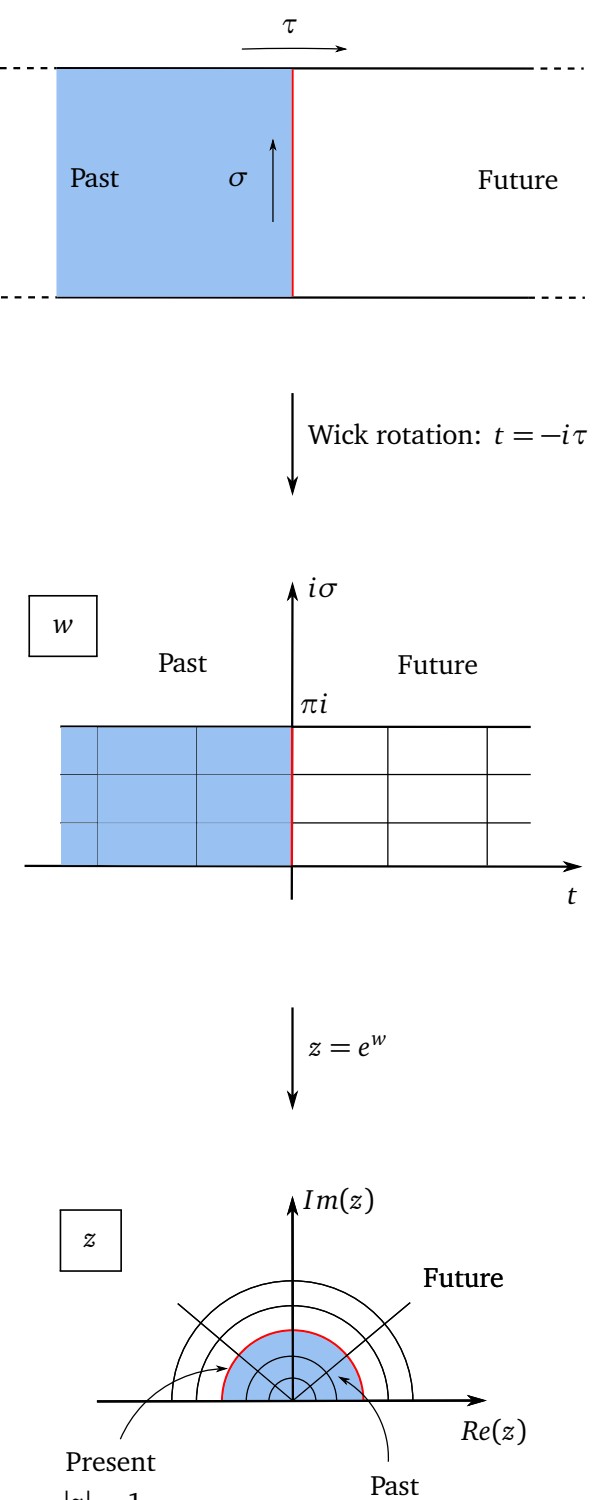

Figure 4.6: The worldsheet of an open string, after a Wick rotation of the time co-ordinate, can be studied as an infinite strip of height $\pi i$ on the Euclidean plane of coordinate $w$. With the exponential function it can then be mapped to the upper half of the complex plane with coordinate $z$, where the ($t = 0$) slice of the WS corre-sponds to the unit semi-circle $|z| = 1$ (marked in red): the past ($t < 0$, marked in blue) lies inside such a semi-circle, while the future ($t > 0$) fills the external region.

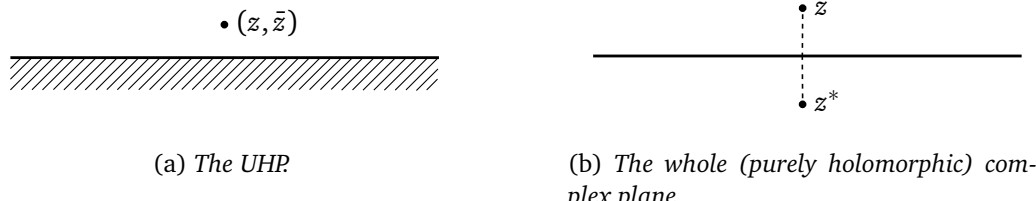

(a) *The UHP.*

(b) *The whole (purely holomorphic) complex plane.*

Figure 4.7: On the left we have the UHP, whose points have a $(z, \bar{z})$ dependence. On the right we have the whole complex plane, having a holomorphic dependence on $z$ and $z^*$.

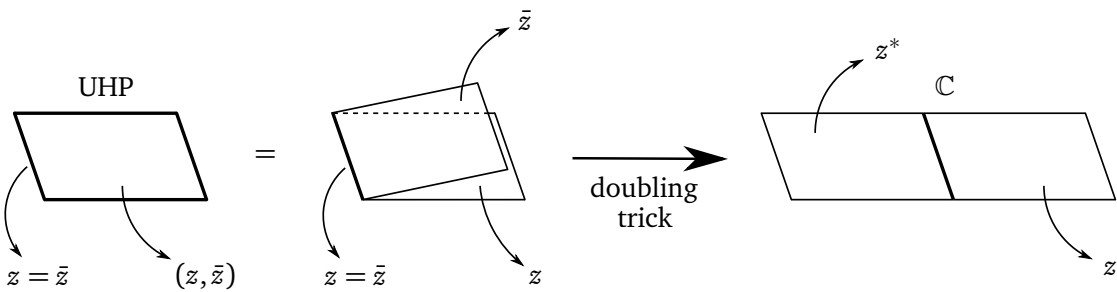

Figure 4.8: The doubling trick.

where the contours of anti-holomorphic coordinates are *clockwise* (while contours of holomorphic currents, as usual, counter-clockwise). This is the same formula we would get in absence of a boundary, except that now $\epsilon(z) = \bar{\epsilon}(\bar{z})$ at $z = \bar{z}$ and analogously for $T$.

In order to proceed in the calculation, we can use the so-called *doubling trick*: instead of considering the UHP whose points have a $(z, \bar{z})$ dependence (see fig. 4.74.7(a)), we can consider the whole complex plane $\mathbb{C}$ with a purely holomorphic dependence on $z$ and $z^*$, being $z^*$ the complex conjugate of $z$ (see fig. 4.74.7(b)).

As shown in fig. 4.8, this procedure "unfolds" the UHP, which has a double dependence on ($z$ and $\bar{z}$) to the whole complex plane, through the boundary $z = \bar{z}$.

With the doubling trick, everything becomes holomorphic (see fig. 4.9).

Therefore we can carry on our calculation as follows:

$$
\begin{aligned}
\delta_{(\epsilon,\bar{\epsilon})}\phi(z,\bar{z}) &= -\oint_z \frac{d\tilde{z}}{2\pi i}\epsilon(\tilde{z})\,\underline{T(\tilde{z})\phi(z,\bar{z})} + \oint_{\bar{z}} \frac{d\bar{\tilde{z}}}{2\pi i}\epsilon(\bar{\tilde{z}})\,\underline{\overline{T}(\bar{\tilde{z}})\phi(z,\bar{z})} \\
&= -\oint_z \frac{d\tilde{z}}{2\pi i}\epsilon(\tilde{z})\,\underline{T(\tilde{z})\phi(z,z^*)} - \oint_{z^*} \frac{d\tilde{z}}{2\pi i}\epsilon(\tilde{z})\,\underline{T(\tilde{z})\phi(z,z^*)} \\
&= -\oint_{z,z^*} \frac{d\tilde{z}}{2\pi i}\epsilon(\tilde{z})T_{\mathbb{C}}(\tilde{z})\phi(z,z^*)\,,
\end{aligned}
\tag{4.207}
$$

where in the last step we deformed the integration path as shown in fig. 4.10.

We have also defined

$$
T_{\mathbb{C}}(\tilde{z}) = \begin{cases} T_{\text{UHP}}(\tilde{z})\,, & \text{for } Im(\tilde{z}) > 0 \text{ (above the boundary)}, \\ \overline{T}_{\text{UHP}}(\bar{\tilde{z}})\,, & \text{for } Im(\tilde{z}) < 0 \text{ (below the boundary)}. \end{cases}
\tag{4.208}
$$

This construction agrees with the boundary conditions (4.202). Notice also that the bulk field $\phi(z, \bar{z})$ is reinterpreted as a *bilocal* holomorphic field $\phi(z, z^*)$. We will see in the following explicit examples that this interpretation *depends on the boundary conditions* and that the same

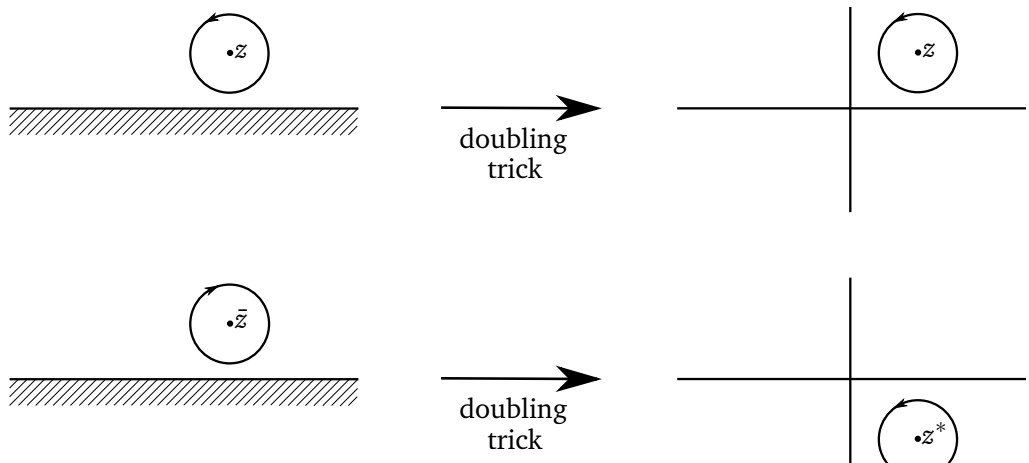

Figure 4.9: The doubling trick makes everything holomorphic.

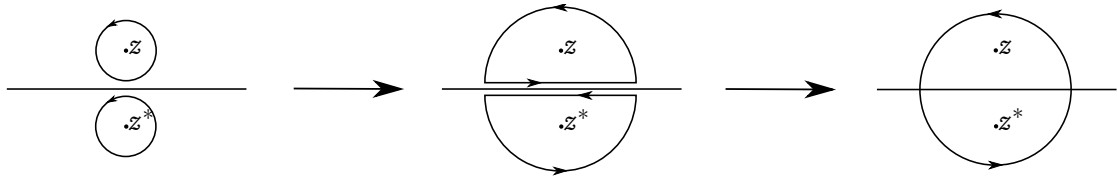

Figure 4.10: Deformation of the integration path performed in the last step of equation (4.207). The vertical axis of the complex plane is not shown for convenience.

bulk field will in general map to different bilocal holomorphic fields depending on the boundary conditions (i.e. the kind of $D$-brane).

### 4.3.2 Matter $X$-BCFT

We are now ready to study the matter $X$-CFT. The action of the $X$ field on the complex plane is given by (recall eq. (4.10))

$$S_{\text{matter}} = \frac{1}{2\pi\alpha'} \int_{\text{WS}} d^2z \, \partial X(z,\bar{z}) \cdot \bar{\partial} X(z,\bar{z}), \tag{4.209}$$

where, as usual, we defined $\partial = \partial_z$ and $\bar{\partial} = \partial_{\bar{z}}$.

Its EOM, being in the open string case, are (3.162). Moreover, in these complex coordinates we can write

$$\boxed{X^\mu(z,\bar{z}) = X_L^\mu(z) + X_R^\mu(\bar{z}),} \tag{4.210}$$

where the two chiral fields $X_L^\mu(z)$ and $X_R^\mu(\bar{z})$ are

$$X_L^\mu(z) = \frac{X_0^\mu}{2} + c^\mu/2 - i\alpha' \frac{P^\mu}{2} \log z + i\sqrt{\frac{\alpha'}{2}} \sum_{n\neq 0} \frac{1}{n} \alpha_n^\mu z^{-n}, \tag{4.211a}$$

$$X_R^\mu(\bar{z}) = \frac{X_0^\mu}{2} - c^\mu/2 - i\alpha' \frac{P^\mu}{2} \log \bar{z} + i\sqrt{\frac{\alpha'}{2}} \sum_{n\neq 0} \frac{1}{n} \alpha_n^\mu \bar{z}^{-n}, \tag{4.211b}$$

where the two fields have only half of the total momentum of the center of mass. The stress-energy tensor of this theory is

$$T(z) = -\frac{1}{\alpha'} :\partial X \cdot \partial X: . \tag{4.212}$$

On the other hand, if we are interested in the current, we may recall that

$$\partial X^\mu(z,\bar{z}) \text{ primary of } h = (1,0) \implies \text{current } j^\mu(z), \tag{4.213a}$$

$$\bar{\partial} X^\mu(z,\bar{z}) \text{ primary of } h = (0,1) \implies \text{current } \bar{j}^\mu(\bar{z}). \tag{4.213b}$$

Depending on the BC, the current will glue in different ways on the boundary:

- Neumann BC along the $\mu$ direction: $j^\mu(z) = \bar{j}^\mu(\bar{z})$ on the boundary $z = \bar{z}$,

- Dirichlet BC along the $i$ direction: $j^i(z) = -\bar{j}^i(\bar{z})$ on the boundary $z = \bar{z}$.

Hence the boundary is labelled by its BC (N or D). More in general, we can write the *gluing condition* on the boundary for the current as

$$\boxed{j^\mu(z) = \Omega_A^{(j)} \bar{j}^\mu(\bar{z}), \quad \text{for } z = \bar{z},} \tag{4.214}$$

with

$$\boxed{\Omega_A = \begin{cases} +1, & \text{for A = Neumann}, \\ -1, & \text{for A = Dirichlet}, \end{cases}} \tag{4.215}$$

which is called *gluing map*.

This reasoning applies in the same way to the $X$ field. Remembering the decomposition (4.210) of $X^\mu(z,\bar{z})$ into the chiral fields $X_L^\mu(z)$ and $X_R^\mu(\bar{z})$, we have that

$$\boxed{X_L^\mu(z) = \Omega_A^{(X)} X_R^\mu(\bar{z}), \quad \text{for } z = \bar{z},} \tag{4.216}$$

where the gluing map is again (4.215). Thus, using the doubling trick the gluing condition becomes

$$X_L^{(N)}(z) \longrightarrow X(z), \qquad\qquad X_R^{(N)}(\bar{z}) \longrightarrow X(z^*), \tag{4.217a}$$

$$X_L^{(D)}(z) \longrightarrow X(z), \qquad\qquad X_R^{(D)}(\bar{z}) \longrightarrow -X(z^*). \tag{4.217b}$$

As said, we know that on the boundary the Virasoro algebra is preserved (thanks to the BC (4.202)). Moreover, in the matter $X$-BCFT there is also an additional symmetry, namely the oscillator algebra, encapsulated by the current algebra, which comes in two copies (one for $j^\mu$ and one for $\bar{j}^\mu$):

$$\underline{j^\mu(z_1) j_\mu(z_2)} = \frac{1}{(z_1 - z_2)^2} \xrightarrow{\text{osc}} \left[\alpha_n^\mu, \alpha_m^\mu\right] = n\delta_{n+m,0}, \tag{4.218a}$$

$$\underline{\bar{j}^\mu(\bar{z}_1) \bar{j}_\mu(\bar{z}_2)} = \frac{1}{(\bar{z}_1 - \bar{z}_2)^2} \xrightarrow{\text{osc}} \left[\tilde{\alpha}_n^\mu, \tilde{\alpha}_m^\mu\right] = n\delta_{n+m,0}, \tag{4.218b}$$

This symmetry is preserved by the gluing condition (4.214) on the boundary, for which we have (4.215)

$$\Omega_A^{(j)} = \pm 1 \implies \left(\Omega_A^{(j)}\right)^2 = 1. \tag{4.219}$$

This result is consistent with the gluing of the stress-energy tensor:

$$\begin{cases} T = \frac{1}{2} :j \cdot j: \\ \overline{T} = \frac{1}{2} :\bar{j} \cdot \bar{j}: \end{cases} \implies \left( \Omega_{\mathrm{A}}^{(j)} \right)^2 = 1 \,. \tag{4.220}$$

Using the doubling trick, we can define (analogously to (4.208))

$$j_{\mathbb{C}}^{\mu}(z) = \begin{cases} j_{\mathrm{UHP}}^{\mu}(z), & \text{for } Im(z) > 0 \ \text{(above the boundary)}, \\ \Omega_{\mathrm{A}}^{(j)} \bar{j}_{\mathrm{UHP}}^{\mu}(\bar{z}), & \text{for } Im(z) < 0 \ \text{(below the boundary)}, \end{cases} \tag{4.221}$$

which is continuous at the boundary.

In a similar way, starting from the chiral fields $\mathcal{V}_{i,\,\mathrm{UHP}}^{(h)}(z)$ and $\overline{\mathcal{V}}_{i,\,\mathrm{UHP}}^{(\bar{h})}(\bar{z})$, we can define

$$\mathcal{V}_{i,\,\mathbb{C}}^{(h)}(z) = \begin{cases} \mathcal{V}_{i,\,\mathrm{UHP}}^{(h)}(z), & \text{for } Im(z) > 0 \ \text{(above the boundary)}, \\ \left( \Omega_{\mathrm{A}}^{(\mathcal{V})} \right)_i^j \overline{\mathcal{V}}_{j,\,\mathrm{UHP}}^{(\bar{h})}(\bar{z}), & \text{for } Im(z) < 0 \ \text{(under the boundary)}, \end{cases} \tag{4.222}$$

where the gluing map is now a matrix $\Omega$ which should preserve the V-V OPE, since we considered the general case where $\mathcal{V}_{i,\,\mathrm{UHP}}^{(h)}(z)$ is a family of operators. If we now consider a bulk field

$$\phi(z,\bar{z}) = \phi_{ij} \mathcal{V}^i(z) \overline{\mathcal{V}}^j(\bar{z}), \tag{4.223}$$

in the presence of a boundary it will become

$$\phi(z,z^*) = \phi_{ij} \left( \Omega_{\mathrm{A}}^{(\mathcal{V})} \right)_k^j \mathcal{V}^i(z) \mathcal{V}^k(z^*). \tag{4.224}$$

For example we can consider the local (non-holomorphic) operator

$$\Phi(z,\bar{z}) = j(z) \cdot \bar{j}(\bar{z}) \ \longrightarrow \ \alpha_{-1} \cdot \tilde{\alpha}_{-1} \left| 0 \right\rangle, \tag{4.225}$$

defining a closed string state. If we place it on the complex plane $\mathbb{C}$ using the doubling trick, it will become a bi-local (completely holomorphic) operator that, depending on the BC, can be written as

$$\Phi^{(\mathrm{N})}(z,z^*) = j(z) \Omega_{\mathrm{N}}^{(j)} j(z^*) = j(z) j(z^*), \tag{4.226a}$$

$$\Phi^{(\mathrm{D})}(z,z^*) = j(z) \Omega_{\mathrm{D}}^{(j)} j(z^*) = -j(z) j(z^*). \tag{4.226b}$$

This is the mechanism that enables closed strings to distinguish different $D(p)$-branes (see exercise 4.3.1).

### 4.3.2.1 Correlation functions of bulk fields

We may first focus on the 1-point functions of a generic field $\phi^{(h,\bar{h})}(z,\bar{z})$. It does not vanish a priori because, due to the boundary, the theory has a scale, namely the distance $y = \frac{1}{2}|z - \bar{z}|$ of the field from the boundary itself (see fig. 4.11). We then have

$$\boxed{ \langle \phi^{(h,\bar{h})}(z,\bar{z}) \rangle^{(a)} = \frac{C_\phi^{(a)}}{|z - \bar{z}|^{2h}} \, \delta_{h,\bar{h}} \,. } \tag{4.227}$$

Moving on towards 2-point function, let us then analyze in detail the bulk correlator of two $X$ fields.

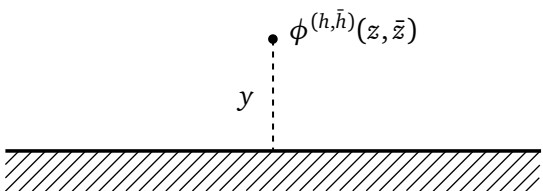

Figure 4.11: A generic field $\phi^{(h,\bar{h})}(z,\bar{z})$ at a distance $y = \frac{1}{2}|z-\bar{z}|$ from the boundary.

Recalling eq. (4.210) and (4.130c), we can write

$$
\begin{aligned}
\langle X(z_1,\bar{z}_1) \cdot X(z_2,\bar{z}_2) \rangle &= \langle X_L(z_1) \cdot X_L(z_2) \rangle + \langle X_R(\bar{z}_1) \cdot X_R(\bar{z}_2) \rangle \\
&= -\frac{\alpha'}{2} \left( \log(z_1-z_2) + \log(\bar{z}_1-\bar{z}_2) \right) \\
&= -\frac{\alpha'}{2} \log\left( |z_1-z_2|^2 \right).
\end{aligned}
\tag{4.228}
$$

In the last step we had to merge the left and right contributions in order to get a single-valued function (and not a multi-valued one, which would be non-local).

We can then consider the same correlation function but on the UHP having Neumann BC. Using the doubling trick and recalling eq. (4.217), we can write

$$
\begin{aligned}
\langle X(z_1,\bar{z}_1) \cdot X(z_2,\bar{z}_2) \rangle^{(\mathrm{N})}_{\mathrm{UHP}} &= \langle \left( X(z_1) + X(z_1^*) \right) \cdot \left( X(z_2) + X(z_2^*) \right) \rangle \\
&= \langle X(z_1) \cdot X(z_2) \rangle + \langle X(z_1) \cdot X(z_2^*) \rangle \\
&\quad + \langle X(z_1^*) \cdot X(z_2) \rangle + \langle X(z_1^*) \cdot X(z_2^*) \rangle \\
&= -\frac{\alpha'}{2} \log\left( |z_1-z_2|^2 \right) - \frac{\alpha'}{2} \left( \log(z_1-z_2^*) + \log(z_1^*-z_2) \right) \\
&= -\frac{\alpha'}{2} \left( \log\left( |z_1-z_2|^2 \right) + \log\left( |z_1-\bar{z}_2|^2 \right) \right).
\end{aligned}
\tag{4.229}
$$

On the other hand, in the Dirichlet case we can write

$$
\langle X(z_1,\bar{z}_1) \cdot X(z_2,\bar{z}_2) \rangle^{(\mathrm{D})}_{\mathrm{UHP}} = -\frac{\alpha'}{2} \left( \log\left( |z_1-z_2|^2 \right) - \log\left( |z_1-\bar{z}_2|^2 \right) \right).
\tag{4.230}
$$

Finally, we can consider the 1-point function. We have that

$$
\langle \partial X \cdot \bar{\partial} X(z,\bar{z}) \rangle^{(\mathrm{N})}_{\mathrm{UHP}} = +\frac{\alpha'}{2} \frac{1}{|z-\bar{z}|^2},
\tag{4.231a}
$$

$$
\langle \partial X \cdot \bar{\partial} X(z,\bar{z}) \rangle^{(\mathrm{D})}_{\mathrm{UHP}} = -\frac{\alpha'}{2} \frac{1}{|z-\bar{z}|^2},
\tag{4.231b}
$$

which is consistent with (4.227).

> **Exercise 4.3.1**
>
> Prove eq. (4.231).

### 4.3.2.2 Plane wave operators

Let us now discuss plane waves operators. We already saw (eq. (4.138)) that the bulk plane wave can be written as

$$
\boxed{\; :e^{iP\cdot X}: (z,\bar{z}) = :e^{iP\cdot X_L}: (z) \, :e^{iP\cdot X_R}: (\bar{z}). \;}
\tag{4.232}
$$

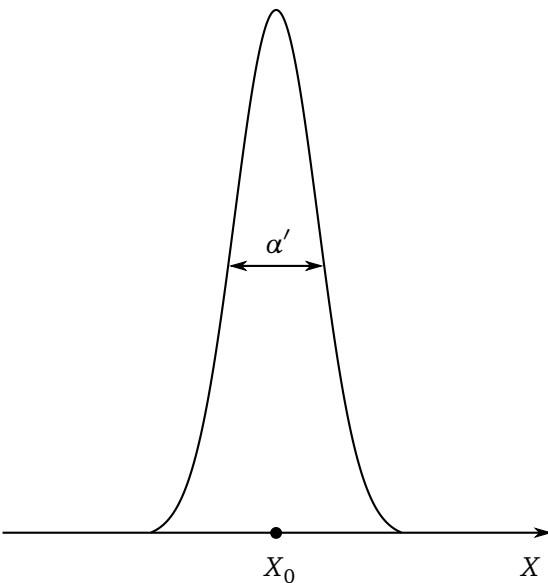

Figure 4.12: A $D(p)$-brane is seen by closed strings as a localized gaussian in the coordinates space, centered in $X = X_0$ and having width of order $\sqrt{\alpha'}$.

On the boundary $z = \bar{z}$, recalling eq. (4.211) and the gluing condition (4.216) for the $X$ field, we realize that the plane wave becomes

- Neumann: $:e^{iP \cdot X}: (z, \bar{z}) = e^{iP \cdot X_L}(z) e^{iP \cdot X_L}(z^*)$,

- Dirichlet: $:e^{iP \cdot X}: (z, \bar{z}) = e^{iP \cdot X_L}(z) e^{-iP \cdot X_L}(z^*)$.

We can then study the 1-point function of such a plane wave as we change the BC. This roughly models the interaction of a closed string with a a $D(p)$-brane. In the Neumann case we have that

$$\langle :e^{iP \cdot X}: (z, \bar{z}) \rangle_{\text{UHP}}^{(\text{N})} = \langle e^{iP \cdot X_L}(z) e^{iP \cdot X_L}(z^*) \rangle = 0, \tag{4.233}$$

since the total momentum of the plane wave cannot be conserved (unless $P = 0$).

On the other hand, in the Dirichlet case (recalling eq. (4.159)) we have that

$$\begin{aligned}
\langle :e^{iP \cdot X}: (z, \bar{z}) \rangle_{\text{UHP}}^{(\text{D})} &= \langle e^{iP \cdot X_L}(z) e^{-iP \cdot X_L}(\bar{z}) \rangle \\
&= (z - \bar{z})^{-\alpha' P^2} \delta(0) \\
&\propto |z - \bar{z}|^{-\alpha' P^2} = e^{-\alpha' \Delta P^2},
\end{aligned} \tag{4.234}$$

where $\Delta = \log|z - \bar{z}|$. In the momentum space, the result we found is a gaussian of width $\frac{1}{\alpha' \Delta}$. If we then perform a Fourier transform we obtain $e^{-X^2/(\alpha' \Delta)}$, which is a gaussian in the coordinates space, centered in $X = 0$ and of width $\alpha' \Delta$, namely having the width of the string. This suggests that the closed string 'sees' the $D(p)$-brane as a localized object at the string scale.

If we now want to consider the more general case where a $D(p)$-brane is centered in a generic $X_0$, we can recall the definition of $X_{DD}$ (eq. (3.181)). Then, in the momentum space we have

$$\langle :e^{iP \cdot X}: (z, \bar{z}) \rangle_{\text{UHP}}^{(\text{D}), X_0} \propto e^{iP \cdot X_0} e^{-\alpha' \Delta P^2}, \tag{4.235}$$

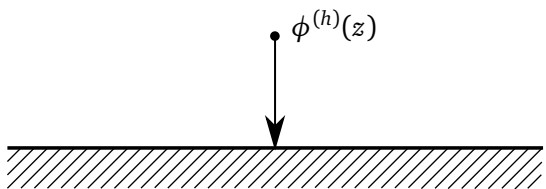

Figure 4.13: A boundary field is defined as a purely holomorphic field laid on the real axis $z = \bar{z}$.

while in the coordinates space we have (see fig. 4.12)

$$\int dP e^{iP \cdot X_0} e^{-\alpha' \Delta P^2} \propto e^{-\frac{(X-X_0)^2}{\alpha' \Delta}}.$$ 

(4.236)

### 4.3.2.3 Boundary fields

So far we discussed fields living in the bulk of the UHP, namely away from the boundary. But as a bulk field $\phi^{(h,\bar{h})}(z, \bar{z})$ approaches the boundary (i.e. $z \to \bar{z}$), it generates a singularity, as we immediately notice from the 1-point function (4.227). Using the doubling trick we can write

$$\phi^{(h)}(z, \bar{z}) = \sum_{i,j} \phi_{i,j} \mathcal{V}_i^{(h)}(z) \bar{\mathcal{V}}_j^{(h)}(\bar{z}) = \sum_{i,j} \phi_{i,j} \frac{\mathcal{V}_i^{(h)}(z)}{\Omega_j^k \mathcal{V}_k^{(h)}(z^*)}.$$

(4.237)

Therefore

$$\mathcal{V}_i^{(h)}(z) \mathcal{V}_k^{(h)}(z^*) \propto \sum_p (z - z^*)^k \left[ \mathcal{V}_i \mathcal{V}_k \right]_p (0),$$

(4.238)

and hence, on the boundary, the two chiral operators $\mathcal{V}_i^{(h)}(z)$ and $\mathcal{V}_i^{(h)}(z^*)$ collide. This tells us that a bulk field that lands on the boundary is the same as two chiral fields crashing on each other.

We thus need a new object, called *boundary field*, which is a field living on the real axis $z = \bar{z}$. It can be thought of as a purely chiral field laid down on the boundary from the bulk (see fig. 4.13):

$$\phi_A^{(h)}(x) = \lim_{z \to x} \phi^{(h)}(z),$$

(4.239)

where A is the label that identifies the boundary condition and $x$ is the coordinate on the real axis. In this way there is no anti-holomorphic counterpart and therefore no collision at the boundary.

Let us now define (see fig. 4.14)

$$\partial_x = \partial_\parallel,$$ 

(4.240a)

$$\partial_y = \partial_\perp.$$ 

(4.240b)

Then we can write

$$\partial = \partial_x - i\partial_y,$$ 

(4.241a)

$$\bar{\partial} = \partial_x + i\partial_y.$$ 

(4.241b)

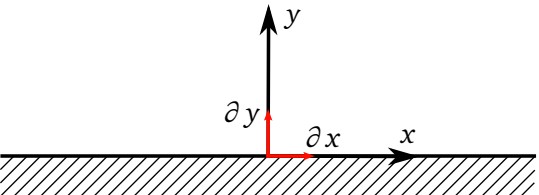

Figure 4.14: In the UHP we can call $x$ the boundary coordinate and $y$ the vertical bulk coordinate. Then we can define the basis vectors to be $\partial x$ and $\partial y$.

We can now consider the free boson along the $\mu$ direction with Neumann BC and recall the current gluing condition:

$$\partial X^\mu(z) = \bar{\partial} X^\mu(\bar{z}), \quad \text{for } z = \bar{z}. \tag{4.242}$$

Using the notation (4.241), we write

$$(\partial - \bar{\partial}) X^\mu(z) \Big|_{z=\bar{z}} = -2i\partial_y X^\mu(z)$$
$$= 0. \tag{4.243}$$

Therefore we can express the Neumann boundary condition as

$$\partial_\perp X^\mu(z) = 0. \tag{4.244}$$

This is exactly what we first wrote in (3.163). Moreover, if we want to build the associated boundary field, we can write

$$\partial X^\mu(x) = \left( \partial_\perp + \partial_\parallel \right) X^\mu(z) \Big|_{z \to x}$$
$$= \partial_\parallel X^\mu(x)$$
$$= \partial_x X^\mu(x). \tag{4.245}$$

On the other hand, the Dirichlet boundary condition for the current

$$\partial X^\mu(z) = -\bar{\partial} X^\mu(\bar{z}) \Big|_{z=\bar{z}} \tag{4.246}$$

can be written as

$$\partial_\parallel X^\mu(z) = 0, \tag{4.247}$$

and the associated boundary field is

$$\partial X^\mu(x) = \left( \partial_\perp + \partial_\parallel \right) X^\mu(z) \Big|_{z \to x}$$
$$= \partial_\perp X^\mu(x)$$
$$= \partial_y X^\mu(x). \tag{4.248}$$

We hence obtained all the boundary fields related to the bosonic current $\partial X$.

Let us then consider the bulk plane wave

$$:e^{iP \cdot X}:(z, \bar{z}) = :e^{iP \cdot X_L}:(z) :e^{iP \cdot X_R}:(\bar{z}). \tag{4.249}$$

We can build a boundary plane wave only if we have Neumann BC, where the momentum is a conserved quantity[24]

$$\left.:e^{2iP \cdot X_L}:(z)\right|_{z \to x} = :e^{2iP \cdot X_L}:(x). \tag{4.250}$$

We may now recall (see eq. (4.211)) that the chiral field $X_L^\mu$ has only half of the total momentum of the center of mass (since $X_R^\mu$ holds the other half). Therefore, in order to have a state with momentum $P$ on the boundary, we have to define

$$X^{\mu \,(\text{open})}(x) = 2X_L^\mu(z)\Big|_{z \to x}, \tag{4.251}$$

which is yet another occurrence of what we have already discussed in the oscillator formalism

$$\alpha_0^{(\text{open})} = 2\alpha_0^{(\text{closed})} = \alpha_0^{(\text{closed})} + \tilde{\alpha}_0^{(\text{closed})}. \tag{4.252}$$

Using eq. (4.211) we can then write

$$X_{(\text{open})}^\mu(x) = X_0^\mu - i\alpha' P^\mu \log(x) + i\sqrt{\frac{\alpha'}{2}} \sum_{n \neq 0} \frac{1}{n} \alpha_n^\mu x^{-n}. \tag{4.253}$$

Given this definition we can finally write the boundary plane wave as

$$:e^{iP \cdot X_{(\text{open})}}:(x) = :e^{2iP \cdot X_L}:(z)\Big|_{z \to x}, \tag{4.254}$$

whose conformal weight is

$$h = 4\frac{\alpha' P^2}{4} = \alpha' P^2. \tag{4.255}$$

We are hence saying that

$$L_0^{(\text{open})} = L_0^{(\text{closed, L sector})} = \alpha' P^2. \tag{4.256}$$

To sum up: we have learned that the chiral field $X_L^\mu$ on its own is non-local, since (recall eq. (4.130c)) it gives rise to non single-valued correlation functions

$$\langle X_L(z_1) \cdot X_L(z_2) \rangle = -\frac{\alpha'}{2} \log(z_1 - z_2). \tag{4.257}$$

However, we have two possibilities to construct a local operator.

- We can merge together the L and R sectors, finding correlation functions whose behaviour is described by an absolute value, which is single-valued. This generates a closed string state, namely an holomorphic+anti-holomorphic bulk field.

- We can put the chiral bulk field $X_L^\mu$ on the boundary, where the real numbers structure generates a natural ordering which removes ambiguities. This generates an open string state, namely a boundary field.

If we then want to give a picture of the Hilbert space of a BCFT, we can divide it into two sectors:

---

[24]We will see in a while that it is not meaningless at all to take a chiral plane wave and put it on a boundary with Dirichlet boundary conditions: this will result in an insertion of a boundary operator that *changes* the value of the modulus of the Dirichlet condition. In general if we take a generic chiral field and put it on the boundary we get a *boundary condition changing operator/field*.

$$x_2 \quad < \quad x_1$$

Figure 4.15: The boundary ordering in the 2-point function case.

- bulk Hilbert space, whose states are given by bulk fields (inserted at the origin of the complex plane) acting on the $SL(2, \mathbb{C})$-invariant vacuum:

$$\phi(z, \bar{z})\Big|_{z=\bar{z}=0} |0\rangle_{SL(2,\mathbb{C})} = \qquad \phi(z, \bar{z})\Big|_{z=\bar{z}=0} \qquad , \qquad (4.258)$$

- boundary Hilbert space, whose states are given by boundary fields (applied at the origin of the UHP) acting on the $SL(2, \mathbb{R})$-invariant vacuum:

$$\phi(x)\Big|_{x=0} |0\rangle_{SL(2,\mathbb{R})} = \qquad \phi(x)\Big|_{x=0} \qquad . \qquad (4.259)$$

#### 4.3.2.4 Correlation functions of boundary fields

Let us first consider the generic 1-point function of a boundary field, which is given by

$$\langle \phi^{(h)}(x) \rangle_{\text{UHP}} = A\delta_{h,0} \,. \qquad (4.260)$$

On the other hand, the generic 2-point function is given by

$$\langle \phi_1^{(h_1)}(x_1)\phi_2^{(h_2)}(x_2) \rangle_{\text{UHP}} = \frac{C}{(x_1 - x_2)^{2h_1}} \delta_{h1,h2} \,, \qquad (4.261)$$

where $x_1 > x_2$, otherwise the correlator has no meaning. The ordering on the boundary is indeed uniquely determined by the real numbers ordering (see fig. 4.15), and boundary fields cannot overstep one another.

This means that $\langle \phi_1^{(h_1)}(x_1)\phi_2^{(h_2)}(x_2) \rangle$ is different than $\langle \phi_2^{(h_2)}(x_1)\phi_1^{(h_1)}(x_2) \rangle$, although in some cases they can coincide.

Finally, the generic 3-point function is

$$\langle \phi_1^{(h_1)}(x_1)\phi_2^{(h_2)}(x_2)\phi_3^{(h_3)}(x_3) \rangle_{\text{UHP}} = \frac{C_{123}}{(x_1 - x_2)^{h_1+h_2-h_3}(x_2 - x_3)^{-h_1+h_2+h_3}(x_1 - x_3)^{h_1-h_2+h_3}} \,. \qquad (4.262)$$

We can indeed map

$$\begin{cases} x_1 & \longrightarrow \Lambda \longrightarrow \infty \,, \\ x_2 & \longrightarrow 1 \,, \\ x_3 & \longrightarrow 0 \,, \end{cases} \qquad (4.263)$$

using the $SL(2, \mathbb{R})$ transformation $f$ and then write

$$\langle \phi_1^{(h_1)}(x_1)\phi_2^{(h_2)}(x_2)\phi_3^{(h_3)}(x_3) \rangle_{\text{UHP}}$$
$$= \langle f \circ \phi_1^{(h_1)}(x_1) f \circ \phi_2^{(h_2)}(x_2) f \circ \phi_3^{(h_3)}(x_3) \rangle_{\text{UHP}}$$

$$= \left[f'(x_1)\right]^{h_1} \left[f'(x_2)\right]^{h_2} \left[f'(x_3)\right]^{h_3} \langle \phi_1^{(h_1)}(\Lambda) \phi_2^{(h_2)}(1) \phi_3^{(h_3)}(0) \rangle_{\text{UHP}}$$

$$\overset{\frac{\Lambda^{2h_1}}{\Lambda^{2h_1}}}{=} \frac{1}{\Lambda^{2h_1}} \left[f'(x_1)\right]^{h_1} \left[f'(x_2)\right]^{h_2} \left[f'(x_3)\right]^{h_3} \langle \mathcal{I} \circ \phi_1^{(h_1)}\left(-\frac{1}{\Lambda}\right) \phi_2^{(h_2)}(1) \phi_3^{(h_3)}(0) \rangle_{\text{UHP}}$$

$$\overset{\Lambda \to \infty}{=} \frac{C_{123}}{(x_1 - x_2)^{h_1 + h_2 - h_3}(x_2 - x_3)^{-h_1 + h_2 + h_3}(x_1 - x_3)^{h_1 - h_2 + h_3}}, \tag{4.264}$$

where we defined

$$C_{123} = \langle \mathcal{I} \circ \phi_1^{(h_1)}(-\frac{1}{\Lambda}) \phi_2^{(h_2)}(1) \phi_3^{(h_3)}(0) \rangle_{\text{UHP}}. \tag{4.265}$$

We can also evaluate the $n$-point function of plane waves:

$$\langle e^{iP_1 \cdot X}(x_1) \cdots e^{iP_n \cdot X}(x_n) \rangle_{\text{UHP}}^{(\text{N})}$$

$$= \langle e^{2iP_1 \cdot X_L}(z_1) \cdots e^{2iP_n \cdot X_L}(z_n) \rangle \Big|_{z_i \to x_i}$$

$$= \prod_{k<l} e^{\langle 2iP_k \cdot X_L(z_k) 2iP_l \cdot X_L(z_l) \rangle} \Big|_{z_i \to x_i} \cdot \delta\left(\sum_{k=1}^{n} P_k\right)$$

$$= \prod_{k<l} e^{-4P_k \cdot P_l \langle X_L(z_k) X_L(z_l) \rangle} \Big|_{z_i \to x_i} \cdot \delta\left(\sum_{k=1}^{n} P_k\right)$$

$$= \prod_{k<l} e^{4P_k \cdot P_l \frac{\alpha'}{2} \log(z_k - z_l)} \Big|_{z_i \to x_i} \cdot \delta\left(\sum_{k=1}^{n} P_k\right)$$

$$= \prod_{k<l} (x_k - x_l) e^{2\alpha' P_k \cdot P_l} \cdot \delta\left(\sum_{k=1}^{n} P_k\right). \tag{4.266}$$

Notice that thanks to the boundary ordering, given by $\prod_{k<l}$ the quantities $(x_k - x_l)e^{2\alpha' P_k \cdot P_l}$ have no phase ambiguities.

### 4.3.2.5 Boundary fields and $D(p)$-branes

Given the $D(p) - D(p)$ branes system shown in fig. 3.6, we can write its Chan-Paton as

$$\begin{pmatrix} |AA\rangle & |AB\rangle \\ |BA\rangle & |BB\rangle \end{pmatrix}, \tag{4.267}$$

where

$$|AA\rangle = \phi^{(AA)}(x=0)|0\rangle = \quad \overset{A \quad \overset{\phi^{(AA)}(0)}{\bullet} \quad A}{\rule{4cm}{0.4pt}} \quad, \tag{4.268a}$$

$$|AB\rangle = \phi^{(AB)}(x=0)|0\rangle = \quad \overset{A \quad \overset{\phi^{(AB)}(0)}{\bullet} \quad B}{\rule{4cm}{0.4pt}} \quad, \tag{4.268b}$$

$$|BA\rangle = \phi^{(BA)}(x=0)|0\rangle = \quad \overset{B \quad \overset{\phi^{(BA)}(0)}{\bullet} \quad A}{\rule{4cm}{0.4pt}} \quad, \tag{4.268c}$$

$$|BB\rangle = \phi^{(BB)}(x=0)|0\rangle = \quad \overset{B \quad \overset{\phi^{(BB)}(0)}{\bullet} \quad B}{\rule{4cm}{0.4pt}} \quad. \tag{4.268d}$$

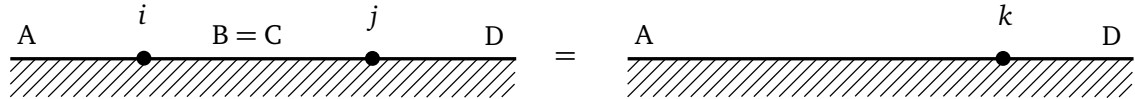

Figure 4.16: The boundary operator algebra.

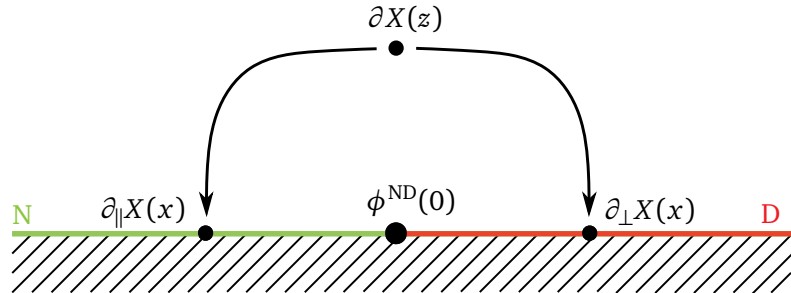

Figure 4.17: The action of the boundary changing field $\phi^{(\mathrm{ND})}(0)$ on the operator $\partial X(z)$.

These $\phi^{(\mathrm{AB})}$ and $\phi^{(\mathrm{BA})}$ are called *boundary condition changing operators*, since they change the BC. They obey the boundary operator algebra:

$$\phi_i^{(\mathrm{AB})}(x)\phi_j^{(\mathrm{CD})}(y) \sim \delta^{\mathrm{BC}} \sum_k C_{ijk}^{\mathrm{ABD}} \phi_k^{(\mathrm{AD})}(y), \qquad (4.269)$$

where the lower indices give the Virasoro representation of the considered field. Notice that the product $\phi_i^{(\mathrm{AB})}(x)\phi_j^{(\mathrm{CD})}(y)$ vanishes unless B = C (see fig. 4.16). This is thus analogous to matrix multiplication.

Let us consider, for example, the field $\phi^{(\mathrm{ND})}(0)$, which changes the BC from Neumann to Dirichlet. If we have $\partial X(z)$ in the bulk and we place it on the boundary, we realize that, if we place it before or after the boundary condition changing operator, it turns into a different boundary field (see fig. 4.17).

Let us now consider two parallel $D(p)$-branes separated along a common Dirichlet direction $y$ at a distance $\Delta_y$, as shown in fig. 4.18. Since the stretched strings have a BC that on the first $D(p)$-brane is given by $X_0 = 0$ while on the second $D(p)$-brane is given by $X_0 = \Delta_y$, there is a change of boundary condition. We can call $\Delta(x)$ the operator responsible for that.

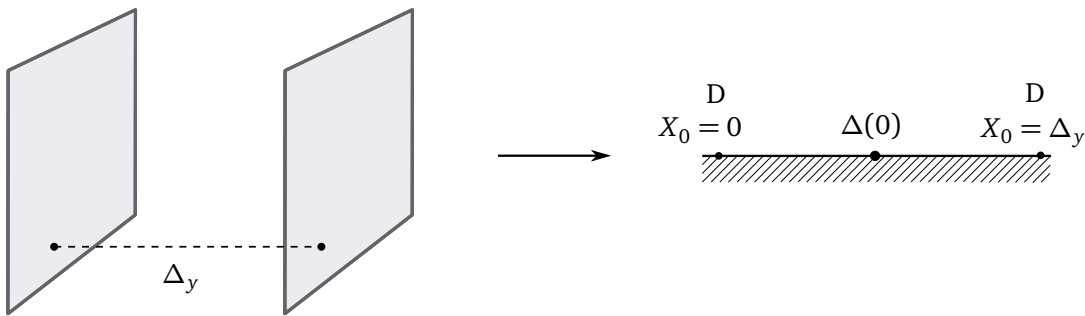

Figure 4.18: Two parallel $D(p)$-branes separated along a common Dirichlet direction $y$ at a distance $\Delta_y$, also shown in the BCFT language through the boundary changing operator $\Delta(x)$ applied in the origin.

We shall then study the $N = 0$ level. Recalling eq. (3.194) and (3.212), we have that the Chan-Paton reads as

$$
\begin{pmatrix} \left| \phi^{11} \right\rangle & \left| \phi^{12} \right\rangle \\ \left| \phi^{21} \right\rangle & \left| \phi^{22} \right\rangle \end{pmatrix},
\tag{4.270}
$$

where the tachyon starting and ending on the same $D(p)$-brane is

$$
\left| \phi^{11} \right\rangle = t(P)e^{iP\cdot X}(0) \left| 0 \right\rangle = t(P) \left| 0, P \right\rangle
\tag{4.271}
$$

(the same holds for $\left| \phi^{22} \right\rangle$), while the stretched tachyon is

$$
\left| \phi^{12} \right\rangle = t(P)e^{iP\cdot X}\Delta(0) \left| 0 \right\rangle = t(P) \left| 0, P \right\rangle^{1,2}
\tag{4.272}
$$

(the same holds for $\left| \phi^{21} \right\rangle$). The field $\Delta(x)$ inserted in the origin is the boundary condition changing operator, which shifts the Dirichlet modulus from 0 (first $D(p)$-brane) to $\Delta_y$ (second $D(p)$-brane). It can be written as a plane wave in the Dirichlet direction $y$, along which the two branes are separated:

$$
\Delta(x) = e^{i\frac{\Delta_y}{\pi\alpha'}X_L^y}(x).
\tag{4.273}
$$

Moreover we have that

$$
T(z)\Delta(x) = \left( \frac{\Delta_y}{2\pi\sqrt{\alpha'}} \right)^2 \frac{\Delta(x)}{(z-x)^2} + \frac{\partial \Delta(x)}{(z-x)} + \text{(regular terms)}.
\tag{4.274}
$$

The stretched tachyon conformal weight is now $h = \alpha'P^2 + \left( \frac{\Delta_y}{2\pi\sqrt{\alpha'}} \right)^2 = 1$, which means $\alpha'm^2 = \left( \frac{\Delta_y}{2\pi\sqrt{\alpha'}} \right)^2 - 1$, exactly as we found previously using oscillators.

### 4.3.3 Ghost $(b, c)$-BCFT and BRST

We can finally briefly build the boundary CFT in the ghost sector. Recalling that $[Q_B, b(z)] = T(z)$ and remembering the gluing condition (4.202) for the stress-energy tensor, we can write the gluing condition for the $b$-ghost as

$$
b(z) = \bar{b}(\bar{z}), \quad \text{for } z = \bar{z}.
\tag{4.275}
$$

Moreover, since we have (see eq. (4.169)) that

$$
b(z_1)c(z_2) = \frac{1}{(z_1 - z_2)} + \text{(regular terms)},
\tag{4.276a}
$$

$$
\bar{b}(\bar{z}_1)\bar{c}(\bar{z}_2) = \frac{1}{(\bar{z}_1 - \bar{z}_2)} + \text{(regular terms)},
\tag{4.276b}
$$

we can write the gluing condition for the $c$-ghost as

$$
c(z) = \bar{c}(\bar{z}), \quad \text{for } z = \bar{z}.
\tag{4.277}
$$

The BRST charge is thus

$$
Q_B = \sum_{n \in \mathbb{Z}} c_n L_{-n}^{(\text{m})} + \frac{1}{2} :c_n L_{-n}^{(\text{gh})}:,
\tag{4.278}
$$

and

$$
Q_B = \bar{Q}_B.
\tag{4.279}
$$

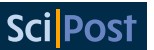

### 4.3.4 Open string physical states

Following the usual logic, open string physical states are elements of the gh=1 open string cohomology, which can be written as

$$c\mathcal{V}(x)\,, \tag{4.280}$$

where $\mathcal{V}$ is a matter boundary primary of weight one. This is the *non integrated* vertex operator. Taking advantage of the fact that $[Q, \mathcal{V}(x)] = \partial(c\mathcal{V})(x)$, we can also construct the integrated vertex operator as

$$\int_{\mathbb{R}} \mathcal{V}(x)\,, \tag{4.281}$$

which, just like the closed string case, is BRST invariant up to possible boundary terms which are vanishing in the case of generic amplitudes.

## 4.4 Open-closed correlation functions

We now want to give a general idea of which are the possible string interactions. The three fundamental interactions are the following.

- CCC: it is the product of two 3-point functions (holomorphic×anti-holomorphic) and is conformally equivalent ($\sim$) to the Riemann sphere with 3 punctures (i.e. insertions of closed string vertex operators), which can also be seen as the the complex plane with 3 fields inserted.

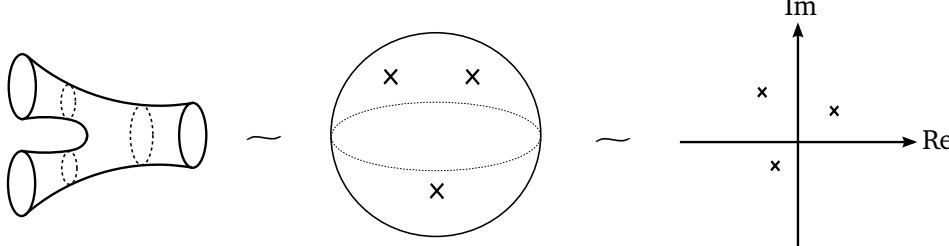

Figure 4.19: CCC correlation function.

- OOO: it is an holomorphic 3-point function and is conformally equivalent to the disk with 3 punctures on the boundary, which can also be seen as the UHP with 3 fields inserted on the real axes.

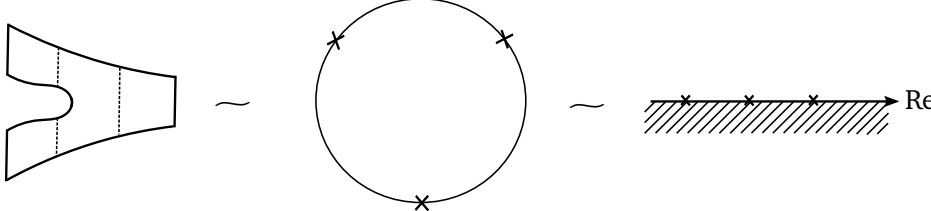

Figure 4.20: OOO correlation function.

- CO: it is an holomorphic 3-point function, where C is bilocal (holomorphic+anti-holomorphic) and is conformally equivalent to the disk with a puncture in the bulk and a puncture on the boundary, which can also be seen as the UHP with one field inserted in the bulk and one on the real axes.

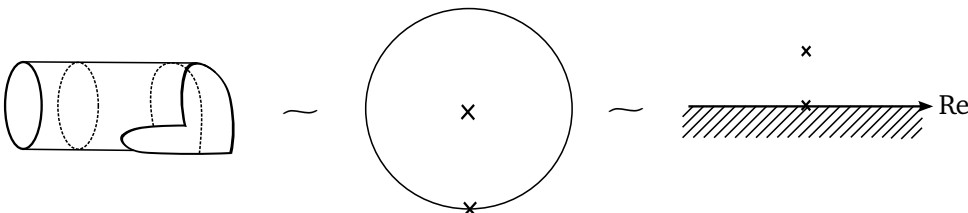

Figure 4.21: CO correlation function.

Given these fundamental ingredients, we can build the following four categories of correlation functions.

- **Closed strings only**

If we write the generic closed string operator as

$$\phi_i(z_i, \bar{z}_i) = \sum_{k_i, l_i} (\Phi_i)_{k_i l_i} \mathcal{V}_{k_i}(z_i) \bar{\mathcal{V}}_{l_i}(\bar{z}_i), \tag{4.282}$$

the generic purely bulk (no boundary) correlation function is given by (see fig. 4.22)

$$\langle \phi_1(z_1, \bar{z}_1) \cdots \phi_n(z_n, \bar{z}_n) \rangle = \prod_{i=1}^{n} \left( \sum_{k_i, l_i} (\Phi_i)_{k_i l_i} \right) \langle \prod_{i=1}^{n} \mathcal{V}_{k_i}(z_i) \rangle \cdot \langle \prod_{i=1}^{n} \bar{\mathcal{V}}_{l_i}(\bar{z}_i) \rangle. \tag{4.283}$$

- **Closed strings with $D(p)$-brane**

If we use the doubling trick to write the generic closed operator as

$$\phi_i(z_i, \bar{z}_i) = \sum_{k_i, l_i} (\Phi_i)_{k_i l_i} \Omega_i \mathcal{V}_{k_i}(z_i) \mathcal{V}_{l_i}(z_i^*), \tag{4.284}$$

where $\Omega_i$ is the gluing map, the generic purely bulk with boundary correlation function is given by (see fig. 4.23)

$$\langle \phi_1(z_1, \bar{z}_1) \cdots \phi_n(z_n, \bar{z}_n) \rangle_{\text{UHP}}^{(A)} = \prod_{i=1}^{n} \left( \sum_{k_i, l_i} (\Phi_i)_{k_i l_i} \Omega_i \right) \langle \prod_{i=1}^{n} \mathcal{V}_{k_i}(z_i) \mathcal{V}_{l_i}(z_i^*) \rangle, \tag{4.285}$$

where A is the boundary condition. The correlation function, thanks to the doubling trick, is completely holomorphic, but it is now a $2n$-point function.

- **Open strings only**

If we write the generic boundary field as

$$\psi_i(x_i) = \Psi_i \mathcal{V}_i(x_i), \tag{4.286}$$

where $\Psi_i$ is a matrix and $\mathcal{V}_i(x_i)$ is a boundary vertex operator, the generic purely boundary (no bulk) correlation function is given by (see fig. 4.24)

$$\langle \psi_1(x_1) \cdots \psi_n(x_n) \rangle_{\text{UHP}}^{(A)} = \text{Tr} \left\{ \prod_{i=1}^{n} \Psi_i \right\} \langle \prod_{i=1}^{n} \mathcal{V}_i(x_i) \rangle. \tag{4.287}$$

- **Open strings and closed strings**

If we consider a process involving open strings and closed strings, the generic mixed (bulk with boundary) correlation function is given by (see fig. 4.25)

$$\langle \psi_1(x_1) \cdots \psi_n(x_n) \phi_1(z_1, \bar{z}_1) \cdots \phi_n(z_n, \bar{z}_n) \rangle$$

$$= \text{Tr} \left\{ \prod_{i=1}^{n} \Psi_i \right\} \prod_{i=1}^{n} \left( \sum_{k_i, l_i} (\Phi_i)_{k_i l_i} \Omega_i \right) \langle \prod_{i=1}^{n} \mathcal{V}_i(x_i) \prod_{i=1}^{n} \mathcal{V}_{k_i}(z_i) \mathcal{V}_{l_i}(z_i^*) \rangle. \tag{4.288}$$

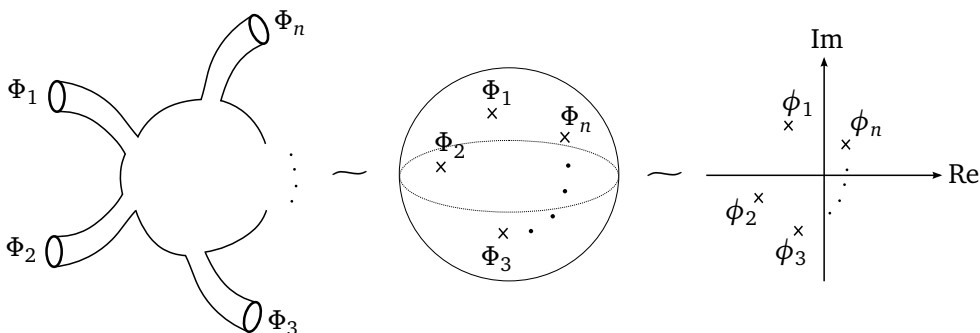

Figure 4.22: A generic correlation function of $n$ closed strings.

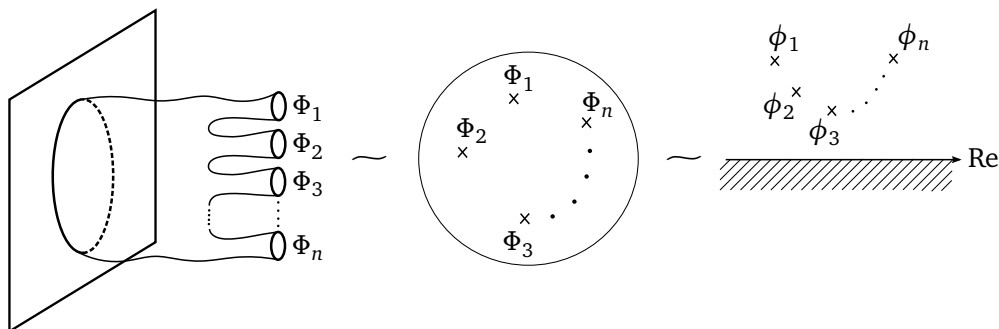

Figure 4.23: A generic correlation function of $n$ closed strings with a $D(p)$-brane.

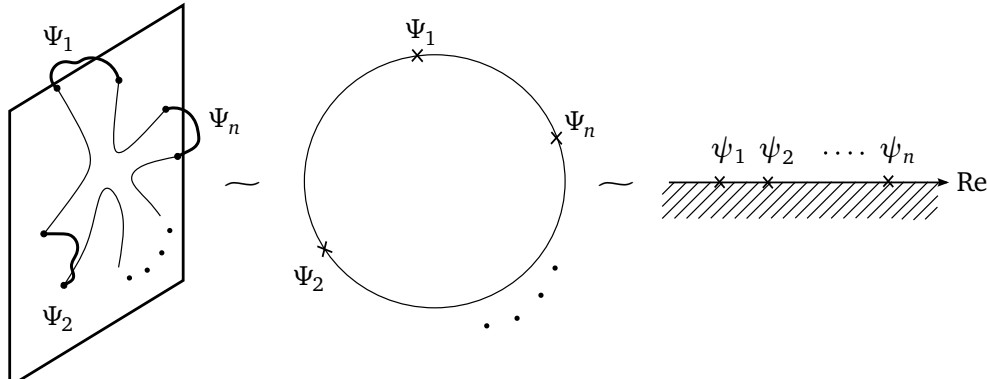

Figure 4.24: A generic correlation function of $n$ open strings.

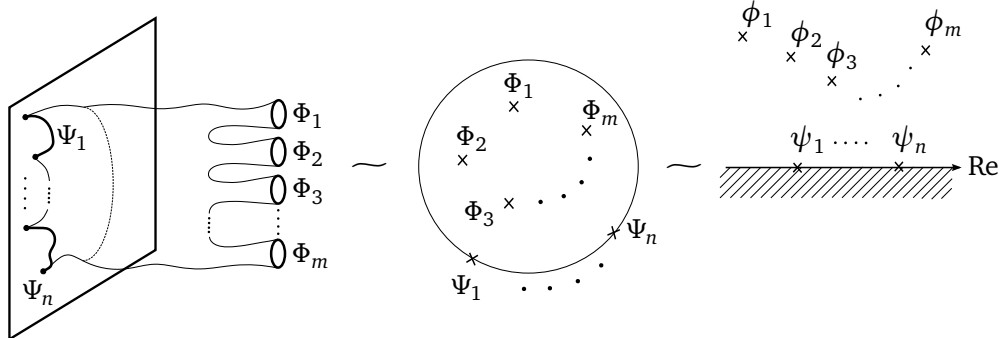

Figure 4.25: A generic correlation function of $n$ open strings and $m$ closed strings.

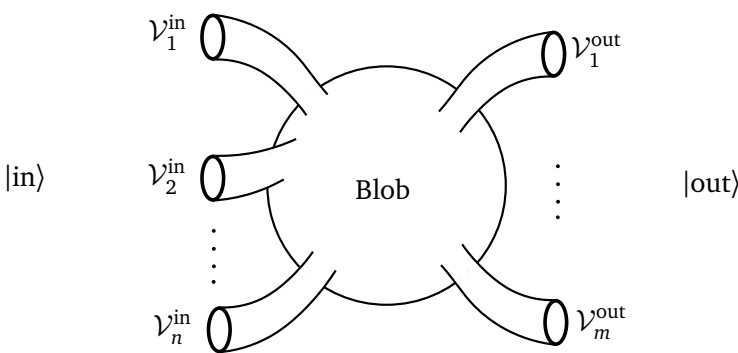

Figure 5.1: Closed strings scattering.

# 5 String perturbation theory

## 5.1 Closed strings amplitudes

We are now ready to compute simple string scattering amplitudes. At first it is necessary to fix the string background that typically will force us to work in a BCFT due to the presence of $D$-branes.

We can start with the simplest case, namely no $D$-branes in the background, just closed strings. Scattering amplitudes are $\mathbb{C}$-numbers giving the probability of transition from an $|\text{in}\rangle$ state to an $|\text{out}\rangle$ state (see fig. 5.1).

Since string theory is constructed from a $2d$ field theory with Diff $\times$ Weyl gauge symmetry, its scattering amplitudes will be naturally associated to correlation functions of gauge invariant operators in the $2d$ theory. The correlation functions will be evaluated using the Polyakov path integral, with insertions of gauge invariant operators. What gauge invariant insertions can produce the string physical states? We already know the answer from the previous chapter, where we saw that the vertex operators $\mathcal{V}$ are gauge invariant by construction and they represent the physical string states. Then the typical $n$-strings amplitude will have the form

$$\mathcal{A}(1,\ldots,n) = \int \frac{\mathcal{D}h_{\alpha\beta}\mathcal{D}X^{\mu}}{\text{Vol}(\mathcal{G})}\,\mathscr{V}_1\cdots\mathscr{V}_n\,e^{-S[h,x]}\,. \tag{5.1}$$

Notice that we are now using an Euclidean path integral in two dimensions, because we have Wick rotated the worldsheet theory to have a standard CFT description. The most natural choice for these $\mathcal{V}_i$ falls back to the integrated vertex operators which in covariant language (before fixing conformal gauge) can be written as

$$\mathscr{V}_i = \int d^2\sigma\,\sqrt{h}\,\mathcal{V}_i(\sigma)\,, \tag{5.2}$$

which are diff-invariant a priori being integrated over the entire worldsheet. As we will see shortly, we will have to sharpen this intuition.

A key aspect in all of this discussion is represented by the metric integral $\mathcal{D}h$. Previously, when we were only interested in the BRST structure of the free string, we were implicilty working on a worldsheet with the topology of a sphere. On a more general ground we have now to understand the integral over metrics to also include all possible two-dimensional topologies. This is indeed the prescription of the path integral: to some over all possible paths (in this case surfaces) connecting the initial and the final configurations, described by the vertex operators

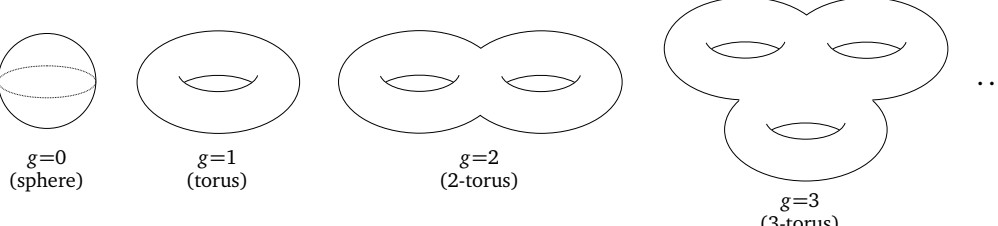

Figure 5.2: The surfaces classification in term of the genus $g$.

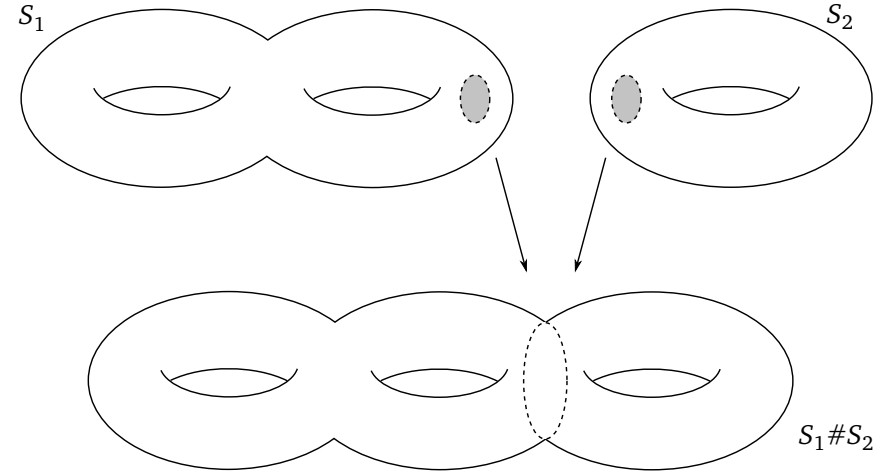

Figure 5.3: The connected sum of two surfaces $S_1$ and $S_2$ is a new surface $S_1 \# S_2$ obtained by deleting a disk from $S_1$ and $S_2$ and gluing together the remaining boundary circles.

$\mathcal{V}_i$. It is therefore more appropriate to explicitly write

$$\int \mathcal{D}h_{\alpha\beta} = \sum_{g \in \text{Top}} \int \mathcal{D}h^{(g)}_{\alpha\beta}, \tag{5.3}$$

where $h^{(g)}_{\alpha\beta}$ is the generic metric (to be integrated over) on the worldsheet with topology $g$. To fully grasp the meaning of this summation we can now use a powerful result of low dimensional topology, the *classification theorem of closed surfaces*, which states that the topology of all orientable surfaces without boundary is classified by the genus $g$, i.e. the number of "handles" (see fig. 5.2).

In other words, a closed surface without boundary and genus $g$ is homeomorphic to a connected sum (see fig. 5.3) of $g$-torii. If $g = 0$, the surface is homeomorphic to a sphere.

To better account these multiple worldsheet topologies, let us remind that on any $d$-dimensional Riemannian manifold it is possible to add to the action a purely metric term, namely the (euclidean) Einstein-Hilbert action:

$$S_{EH} = \frac{1}{2^d \pi} \int d^d x \sqrt{h} R^{(d)}. \tag{5.4}$$

Here the metric can be interpreted as a dynamical local field with a number of massless propagating degrees of freedom (gravitons) equal to

$$\#\text{d.o.f.} = \frac{d(d-3)}{2}, \quad d \geq 3. \tag{5.5}$$

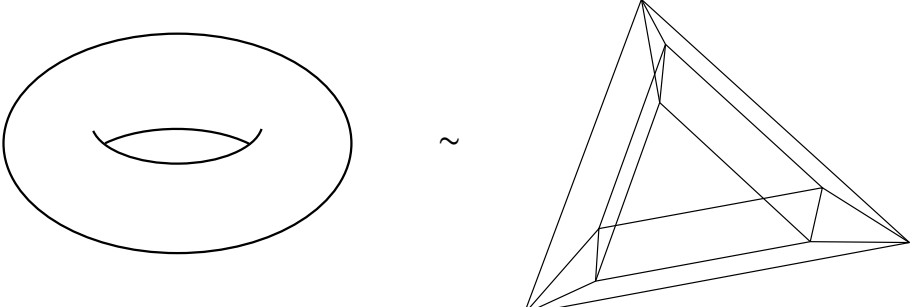

Figure 5.4: The torus and an homeomorphic polyhedron.

It is clear then that for $d \leq 3$ we do not get any local degrees of freedom. In particular in $d = 2$ the above action becomes a topological invariant. The Gauss-Bonnet theorem states that for a closed surface without boundary we have

$$S_{EH} = \frac{1}{4\pi} \int d^2z \sqrt{h} R^{(2)} = \chi, \qquad (5.6)$$

where $\chi$ is the *Euler characteristic* of the surface. Given a polyhedron we have

$$\chi = \text{Vertices} - \text{Edges} + \text{Faces}. \qquad (5.7)$$

Therefore for a generic surface $S$ the Euler number can be computed considering any polyhedron homeomorphic to the surface $S$. It is possible to prove that this computation yields

$$\chi = 2 - 2g, \qquad (5.8)$$

with $g$ the genus of S. See fig. 5.4 for the example of the torus and an homeomorphic polyhedron. Now, being $S_{EH}$ a topological term, it is also Diff × Weyl invariant and we can freely add it to the Polyakov action:[25]

$$S_T[h, X] = -\frac{1}{4\pi\alpha} \int d^2z \sqrt{-h}\, h^{\alpha\beta} \partial_\alpha X_\mu \partial_\beta X^\mu + \frac{\lambda}{4\pi} \underbrace{\int d^2z \sqrt{-h} R^{(2)}}_{=8\pi(1-g)}. \qquad (5.9)$$

This addition has an important effect in the path integral, since

$$\mathcal{Z} = \sum_g e^{-2\lambda(1-g)} \int \frac{\mathcal{D}h_{\alpha\beta}^{(g)} \mathcal{D}X^\mu}{\text{Vol}(\mathcal{G})} e^{-S_{\text{Pol}}[h,X]}$$

$$= \sum_g (e^\lambda)^{2(g-1)} \int \frac{\mathcal{D}h_{\alpha\beta}^{(g)} \mathcal{D}X^\mu}{\text{Vol}(\mathcal{G})} e^{-S_{\text{Pol}}[h,X]}. \qquad (5.10)$$

Indeed we see that the summation over all the topologies is transforming into a sort of loop expansion (see fig. 5.5) in terms of the *string coupling constant*

$$g_s = e^\lambda,$$

with a loop counting parameter represented by genus of the Riemann surface. Notice that at this level it seems that this new constant $\lambda$ is yet a new free parameter of string theory (together with $\alpha'$). On the contrary we will later see that $\lambda$ is nothing but the expectation value of the dilaton field $\phi$ and that the string dynamics is also fixing its own coupling constant.

---

[25]There is a more physical reason to do such a thing and we will explore it later.

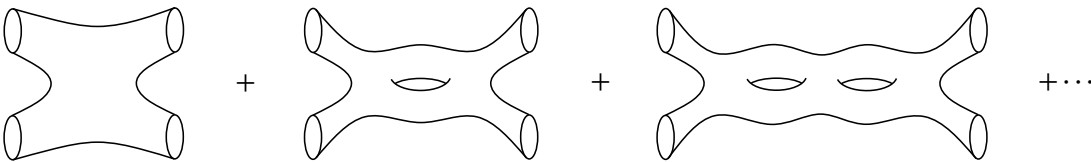

Figure 5.5: The loop expansion.

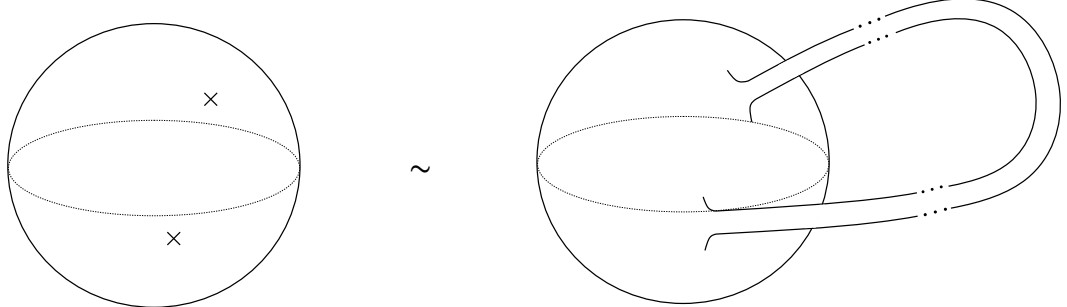

Figure 5.6: A sphere with two punctures has the same Euler number of a sphere with an infinite handle.

What happens now if we insert the vertex operators to compute the $n$-point amplitude $\mathcal{A}(1,\ldots,n)$? Each vertex insertion appears as a puncture on the worldsheet, which therefore becomes a punctured Riemann surface. This changes the topology so we expect to see a change in the Euler characteristic.

To understand if this intuition is correct we take as an example a simple two-point function on a sphere($g = 0$). Intuitively we can think of the two punctures as two semi-infinite tubes. We can further imagine that these two infinite tubes meet and join at infinity thus forming a degenerate handle (see fig. 5.6) as this does not change the curvature.

In this configuration the Euler characteristics shifts from $\chi = 2$ to $\chi = 0$. This allows for an easy generalization: for any Riemann surface with characteristic $\chi_0$, the insertion of $2n$ punctures will be the same (as far as $\chi$ is concerned) as the insertion of $n$ infinite handles. Therefore the punctured Riemann surface has $\chi = \chi_0 - 2n$. This implies that for any vertex operator the Euler characteristic is reduced by 1 and we get the general formula

$$\chi(\Sigma_{g,n}) = 2(1-g) - n,$$ (5.11)

for a surface $\Sigma_{g,n}$ with genus $g$ and $n$ punctures.

The same result can be obtained in a more geometrical way noticing that a puncture $z^*$ can be interpreted as a thin hyperbolic spike centered in the puncture and extending to infinity (see fig. 5.7) which has the geometric effect of shifting the Ricci scalar by a negative delta-function term so that

$$\chi_0 \longrightarrow \chi = \chi_0 - \int d^2z \sqrt{-h}\delta^{(2)}(z - z^*) = \chi_0 - 1,$$ (5.12)

which coincides with the previous result.

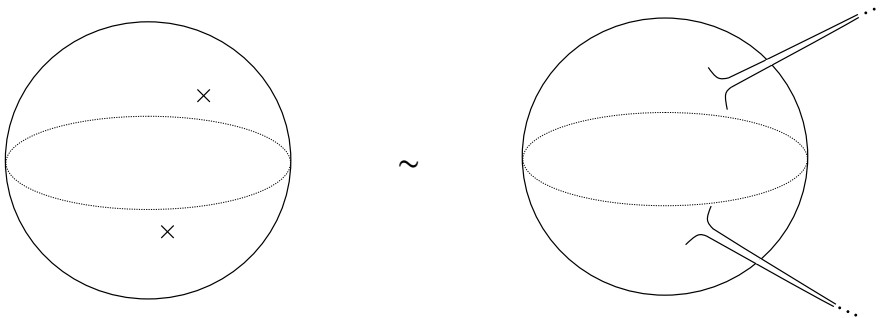

Figure 5.7: A sphere with two punctures is homeomorphic to a sphere with two infinite spikes centered in the punctures.

The physical effect of what we just said is that any $n$ point function is additionally suppressed by a factor of $g_s^n$ with respect the vacuum bubble (0-point amplitude):

$$\mathcal{A}(1,\ldots,n) = g_s^n \sum_{g=0}^{\infty} g_s^{2(g-1)} \int \frac{\mathcal{D}h^g \mathcal{D}X}{\text{Vol}(\mathcal{G})} \, \mathcal{V}_1 \cdots \mathcal{V}_n \, e^{-S_{\text{Pol}}[h,X]}$$

$$= g_s^n \sum_{g=0}^{\infty} g_s^{2(g-1)} \langle\!\langle \mathcal{V}_1 \cdots \mathcal{V}_n \rangle\!\rangle^{(g)}. \tag{5.13}$$

This is the most general form of closed string scattering amplitude: a correlation function of gauge invariant operators in a $2d$ quantum gravity!

### 5.1.1 Tree level amplitudes

Let's now analyze the first order of the perturbative series, genus zero. This is the tree-level contribution. The $\langle\!\langle \mathcal{V}_1 \ldots \mathcal{V}_n \rangle\!\rangle^{(0)}$ is now an integrated correlation function on the complex plane, which we have studied in detail in the previous chapter

$$\langle\!\langle \mathcal{V}_1 \cdots \mathcal{V}_n \rangle\!\rangle^{(0)} = \int \frac{\mathcal{D}h^g \mathcal{D}X}{\text{Vol}(\mathcal{G})} \int d^2z_1 \sqrt{h} \cdots d^2z_n \sqrt{h} \, \mathcal{V}_1 \cdots \mathcal{V}_n \, e^{-S_{\text{Pol}}[h,X]}$$

$$= \int \mathcal{D}X \mathcal{D}c \mathcal{D}b \, d^2z_1 \ldots d^2z_n \, \mathcal{V}_1(z_1,\bar{z}_1) \cdots \mathcal{V}_n(z_n,\bar{z}_n), e^{-S[X,b,c]}$$

$$= \int d^2z_1 \cdots d^2z_n \underbrace{\langle \mathcal{V}_1(z_1,\bar{z}_1) \cdots \mathcal{V}_n(z_n,\bar{z}_n)\rangle}_{\text{CFT correlator on } \mathbb{C}}$$

$$= \mathcal{A}_0 \,, \tag{5.14}$$

where in the second step we gauge-fixed the metric to $h_{\alpha\beta} = \delta_{\alpha\beta}$.

Now we must be careful because we are integrating over a CFT correlator which is invariant under the action of $SL(2,\mathbb{C})$. This means that inside $\langle \mathcal{V}_1(z_1,\bar{z}_1) \ldots \mathcal{V}_n(z_n,\bar{z}_n)\rangle$ the integral over $n$ variables will necessarily give a divergent result, proportional to the infinite volume of $SL(2,\mathbb{C})$. On the other hand it seems that we can just put this amplitude to zero because $\langle \mathcal{V}_1(z_1,\bar{z}_1) \ldots \mathcal{V}_n(z_n,\bar{z}_n)\rangle$ is the correlator of a ghost $\otimes$ matter CFT, and because the $\mathcal{V}_i$ are all in the matter sector, in the ghost sector we are left with

$$\langle 0|1|0 \rangle = \langle 0|0 \rangle = 0 \,. \tag{5.15}$$

We know indeed that the non vanishing correlator is not $\langle 1 \rangle$, but instead $\frac{1}{2}\langle \partial^2 c\, \partial c\, c \rangle = \langle c_{-1} c_0 c_1 \rangle = 1$ and analogously for the anti-holomorphic sector. We are then getting a sort of undetermined product

$$\mathcal{A}_0 = 0 \times \infty\,,$$

so it is clear that we must be very careful with this computation.

Let us start to better understand where the limit $\mathcal{A}_0 \to 0$ is coming from by analyzing explicitly the ghost part of the path integral:

$$\int \mathcal{D}b\mathcal{D}\bar{b}\mathcal{D}c\mathcal{D}\bar{c}\,(1)\,e^{-\frac{1}{2\pi}\int d^2z(b\bar{\partial}c+\bar{b}\partial\bar{c})}\,. \tag{5.16}$$

At first we notice that the action in the exponential does not depend on the zero modes of $c$ and $\bar{c}$, namely the solutions of the equations $\partial\bar{c}=0$, $\bar{\partial}c=0$ which extend globally on the complex plane. Focusing on $c$ it is obvious that the zero modes do not depend on the anti-holomorphic coordinate and in general can be written as a series expansion around 0:

$$c^{(0)}(z) = \sum_{n\leq 1} c_n z^{-n+1}\,, \tag{5.17}$$

where we imposed $c_{n>1}=0$ to assure regularity in 0. However this is not the end of the story, because for this solution to extend correctly over all the complex plane we must also make sure that the solution is regualar at infinity. To do so we perform a coordinate change $z \to -1/z$ so that the zero mode transforms as

$$\mathcal{I} \circ c^{(0)}(z) = \left(\frac{1}{z^2}\right)^{-1} \sum_{n\leq 1} c_n (-1)^n z^{n-1} = \sum_{n\leq 1} c_n (-1)^n z^{n+1}\,, \tag{5.18}$$

which is regular at $z=0$ only if $c_{n<-1}=0$. We are thus left with a simple polynomial solution:

$$c^{(0)}(z) = c_1 + c_0 z + c_{-1} z^2\,, \tag{5.19}$$

with $c^{(0)}$ parameterized by just the three zero modes coefficient $(c_1, c_0, c_{-1})$. By the fact that this solution extends globally on $\mathbb{C}$, every other possible field $c$ can then be rewritten in the form $c(z) = c^{(0)}(z) + C(z)$ with $c^{(0)}$ an arbitrary zero mode, and the integration measure takes the form $\mathcal{D}c = \mathcal{D}C\mathcal{D}c^{(0)} = \mathcal{D}C\, dc_{-1}dc_0dc_1$. The total path integral measure then generally factorizes as $\mathcal{D}c\mathcal{D}\bar{c} = \mathcal{D}C\mathcal{D}\bar{C}\,|dc_{-1}dc_0dc_1|^2$ and the origin of the indeterminate value of the amplitude $\mathcal{A}_0$ becomes clear

$$\mathcal{A}_0 = \underbrace{\int d^2z_1\cdots d^2z_n}_{=\infty} \underbrace{\int \prod_{i=0,\pm 1} dc_i d\bar{c}_i}_{=0} \underbrace{\langle \mathcal{V}_1\cdots\mathcal{V}_n\rangle'}_{\text{no zero modes}}\,, \tag{5.20}$$

since the Berezin integral in $dc_i$ is identically zero given that $\langle \mathcal{V}_1\ldots\mathcal{V}_n\rangle'$ is independent of $c_i$.

On these grounds it is clear then that the easiest way to make this integral finite is by changing the vertex operators inside the correlator. Indeed we know that there are two types of vertex operators, the integrated and the non-integrated vertex operators, which are given by $c\bar{c}\mathcal{V}(z,\bar{z})$, and are BRST invariant. Hence if we consider in the amplitude the insertions of three non-integrated operators the ambiguity will be solved if we write

$$\mathcal{A}_0 = \int d^2z_4\cdots d^2z_n\,\langle c\bar{c}\mathcal{V}_1(z_1,\bar{z}_1)\cdots c\bar{c}\mathcal{V}_3(z_3,\bar{z}_3)\,\mathcal{V}_4(z_4,\bar{z}_4)\cdots\mathcal{V}_n(z_n,\bar{z}_n)\rangle\,. \tag{5.21}$$

The $SL(2,\mathbb{C})$ gauge invariance is now fixed because the three non-integrated vertex operators are fixing three points on the sphere. Moreover the three ghost insertions saturate the zero mode integrals giving a non-vanishing result. In particular, thanks to the splitting of the CFT into ghost and matter sectors, we can separately compute the ghost correlator and obtain

$$\langle c\bar{c}(z_1)c\bar{c}(z_2)c\bar{c}(z_3)\rangle = \underbrace{\langle 0|\, c_{-1}\bar{c}_{-1}c_0\bar{c}_0c_1\bar{c}_1\,|0\rangle}_{=1} \begin{vmatrix} 1 & 1 & 1 \\ z_1 & z_2 & z_3 \\ z_1^2 & z_2^2 & z_3^2 \end{vmatrix}^2 = |z_{12}z_{13}z_{23}|^2\,, \tag{5.22}$$

where we used (4.182) and we have fixed the zero mode convention for a non vanishing inner product as above.

It is interesting to notice that the same result can be obtained by treating the $SL(2,\mathbb{C})$ invariance in the starting $\mathcal{A}_0$ as a gauge invariance to fix using the Faddeev-Popov method. As a matter of fact, let us consider the initial $\mathcal{A}_0$ normalized by the volume of $SL(2,\mathbb{C})$:

$$\mathcal{A}_0 = \int \frac{d^2z_1\cdots d^2z_n}{\mathrm{Vol}(SL(2,\mathbb{C}))}\,\langle \mathcal{V}_1(z_1,\bar{z}_1)\cdots \mathcal{V}_n(z_n,\bar{z}_n)\rangle\,. \tag{5.23}$$

We can apply the Faddeev-Popov procedure by fixing three points $z_i = \hat{z}_i$, $\bar{z}_i = \bar{\hat{z}}_i$, $i = 1,2,3$. The gauge fixing functions will then be $f_i(z_i) = z_i - \hat{z}_i$, $\bar{f}_i(z_i) = \bar{z}_i - \bar{\hat{z}}_i$. Therefore, given the $SL(2,\mathbb{C})$ action

$$\delta_\epsilon z = \epsilon_1 + \epsilon_2 z + \epsilon_3 z^2\,, \qquad \delta_\epsilon \bar{z} = \bar{\epsilon}_1 + \bar{\epsilon}_2\bar{z} + \bar{\epsilon}_3\bar{z}^2\,, \tag{5.24}$$

the usual resolution of the identity takes the form

$$1 = \int |d\epsilon_1 d\epsilon_2 d\epsilon_3|^2 |\delta(\epsilon_1)\delta(\epsilon_2)\delta(\epsilon_3)|^2 = \int [d\epsilon]\prod_{i=1}^{3}|\delta(z_i-\hat{z}_i)|^2\left|\frac{\delta z_i}{\delta\epsilon_j}\right|$$

$$= \int [d\epsilon]\prod_{i=1}^{3}|\delta(z_i-\hat{z}_i)|^2 \begin{vmatrix} 1 & 1 & 1 \\ \hat{z}_1 & \hat{z}_2 & \hat{z}_3 \\ \hat{z}_1^2 & \hat{z}_2^2 & \hat{z}_3^2 \end{vmatrix}^2 = \int [d\epsilon]\prod_{i=1}^{3}|\delta(z_i-\hat{z}_i)|^2|z_{12}z_{13}z_{23}|^2\,, \tag{5.25}$$

so that

$$\mathcal{A}_0 = \underbrace{\left(\int [d\epsilon]\frac{1}{\mathrm{Vol}(SL(2,\mathbb{C}))}\right)}_{=1}\int d^2z_1\cdots d^2z_n\prod_{i=1}^{3}|\delta(z_i-\hat{z}_i)|^2|z_{12}z_{13}z_{23}|^2\langle\mathcal{V}_1\cdots\mathcal{V}_n\rangle$$

$$= \int d^2z_4\cdots d^2z_n\,|\hat{z}_{12}\hat{z}_{13}\hat{z}_{23}|^2\langle\mathcal{V}_1(\hat{z}_1,\bar{\hat{z}}_1)\cdots\mathcal{V}_3(\hat{z}_3,\bar{\hat{z}}_3)\mathcal{V}_4(z_4,\bar{z}_4)\cdots\mathcal{V}_n(z_n,\bar{z}_n)\rangle\,, \tag{5.26}$$

which is exactly the result obtained above. The ghost correlator can then be exaclty identified as the $SL(2,\mathbb{C})$ FP determinant.

Looking at the obtained result, however, it superficially seems that it depends on the position of the three points $\hat{z}_i$, $i = 1,\ldots,3$, which clearly would be inconsistent. We will now prove that this dependence is just apparent. Let us indeed suppose to take two different points $\hat{z}_1, \hat{w}_1$ and two amplitudes $\mathcal{A}_0$ and $\mathcal{A}_0'$ computed respectively in $\hat{z}_1$ and $\hat{w}_1$. Then we can consider the difference of the two amplitudes and notice that this difference has again the form of an amplitude, but containing the difference of the unintegrated vertex operator evaluated at the two points:

$$\mathcal{A}_0 - \mathcal{A}_0' \to \left[c\bar{c}\mathcal{V}(\hat{z}_1,\bar{\hat{z}}_1) - c\bar{c}\mathcal{V}(\hat{w}_1,\bar{\hat{w}}_1)\right]\,. \tag{5.27}$$

By definition of vertex operator we can write

$$c\bar{c}\mathcal{V}(\hat{z}_1,\bar{\hat{z}}_1) - c\bar{c}\mathcal{V}(\hat{w}_1,\bar{\hat{w}}_1) = \sum_{ij}\Phi_{ij}\left[c\mathcal{V}_i(\hat{z}_1)\bar{c}\overline{\mathcal{V}}_j(\bar{\hat{z}}_1) - c\mathcal{V}_i(\hat{w}_1)\bar{c}\overline{\mathcal{V}}_j(\bar{\hat{w}}_1)\right]\,. \tag{5.28}$$

SciPost Phys. Lect. Notes 90 (2025)

Using then eq. (4.195) we obtain:

$$c\mathcal{V}_i(\hat{z}) = c\mathcal{V}_i(\hat{w}) - \int_{\hat{z}}^{\hat{w}} d\xi \, [Q, \mathcal{V}_i(\xi)] \,, \tag{5.29a}$$

$$\bar{c}\overline{\mathcal{V}}_i(\hat{\bar{z}}) = \bar{c}\overline{\mathcal{V}}_i(\hat{w}) - \int_{\hat{\bar{z}}}^{\hat{\bar{w}}} d\bar{\xi} \, [\bar{Q}, \overline{\mathcal{V}}_i(\bar{\xi})] \,, \tag{5.29b}$$

which plugged back into (5.28) allows to write

$$\left[ c\bar{c}\mathcal{V}(\hat{z}_1, \hat{\bar{z}}_1) - c\bar{c}\mathcal{V}(\hat{w}_1, \hat{\bar{w}}_1) \right]$$

$$= \sum_{ij} \Phi_{ij} \left\{ -\left[ Q, \int_{\hat{z}}^{\hat{w}} d\xi \, \mathcal{V}_i(\xi) \right] \bar{c}\, \overline{\mathcal{V}}_j(\bar{w}) - c\, \mathcal{V}_i(\bar{w}) \left[ \bar{Q}, \int_{\hat{\bar{z}}}^{\hat{\bar{w}}} d\bar{\xi} \, \overline{\mathcal{V}}_j(\bar{\xi}) \right] \right.$$

$$\left. + \left[ Q\bar{Q}, \int_{\hat{z}}^{\hat{w}} d\xi \int_{\hat{\bar{z}}}^{\hat{\bar{w}}} d\bar{\xi} \, \mathcal{V}_i(\xi)\overline{\mathcal{V}}_j(\bar{\xi}) \right] \right\}$$

$$= \sum_{ij} \left\{ \left[ Q, A_{ij} \right] + \left[ \bar{Q}, B_{ij} \right] + \left[ Q\bar{Q}, C_{ij} \right] \right\} \,, \tag{5.30}$$

which is BRST-exact. Plugging this back in the complete form of $\mathcal{A}_0 - \mathcal{A}_0'$ we get

$$\mathcal{A}_0 - \mathcal{A}_0' = \int d^2z_4 \cdots d^2z_n \, \langle [Q_B, \ldots] \mathcal{V}_2(\hat{z}_2, \hat{\bar{z}}_2)\mathcal{V}_3(\hat{z}_3, \hat{\bar{z}}_3)\mathcal{V}_4 \cdots \mathcal{V}_n \rangle = 0 \,, \tag{5.31}$$

because

$$Q_B c\bar{c}\mathcal{V}(z, \bar{z}) = 0 \,, \tag{5.32}$$

$$Q_B \int dz d\bar{z} \mathcal{V}(z, \bar{z}) = 0 \quad \text{(up to boundary terms)}. \tag{5.33}$$

For generic external momenta the possible contributions from boundary terms are vanishing and so everything is ok. Repeating the same steps for $z_2$ and $z_3$ it is immediate to see that the amplitude does not depend on any of the three coordinates.

### 5.1.2 Example: Shapiro–Virasoro amplitude

Let's start with the simpler three-tachyons closed string amplitude. In this case we have to fix all of the three vertex operators and there is no integration:

$$\mathcal{A}(p_1, p_2, p_3) = g_s \, t_1(p_1)t_2(p_2)t_3(p_3) \left\langle c\bar{c}e^{ip_1 \cdot X}(z_1, \bar{z}_1) \, c\bar{c}e^{ip_2 \cdot X}(z_2, \bar{z}_2) \, c\bar{c}e^{ip_3 \cdot X}(z_3, \bar{z}_3) \right\rangle$$

$$= g_s \, t_1(p_1)t_2(p_2)t_3(p_3)\delta(p_1 + p_2 + p_3) \,. \tag{5.34}$$

Notice that this is a 3-point function of weight-zero primary bulk fields. Therefore it is a constant and the constant is given by the momentum conserving delta function. The first non trivial amplitude is the 4-tachyons scattering. In this case we have to fix 3 vertex operators and we have to integrate the 4th over the whole complex plane:

$$\mathcal{A}(p_1, p_2, p_3, p_4) = g_s^2 \, t_1(p_1)t_2(p_2)t_3(p_3)t_4(p_4)$$

$$\times \int d^2z \, \left\langle c\bar{c}e^{ip_1 \cdot X}(z_1, \bar{z}_1) \, c\bar{c}e^{ip_2 \cdot X}(z_2, \bar{z}_2) \, c\bar{c}e^{ip_3 \cdot X}(z_3, \bar{z}_3)e^{ip_4 \cdot X}(z, \bar{z}) \right\rangle . \tag{5.35}$$

It is common to choose the fixed positions as $z_1 = \Lambda \to \infty$, $z_2 = 1$ and $z_3 = 0$. The ghost correlator is given by

$$\left\langle c\bar{c}(\Lambda, \bar{\Lambda})\, c\bar{c}(1, \bar{1})\, c\bar{c}(0, \bar{0}) \right\rangle = |(\Lambda - 1)\Lambda|^2 \sim |\Lambda|^4, \tag{5.36}$$

while, using (4.159), the matter four-point function gives

$$\begin{aligned}
&\left\langle e^{ip_1 \cdot X}(\Lambda, \bar{\Lambda})\, e^{ip_2 \cdot X}(1, \bar{1})\, e^{ip_3 \cdot X}(0, \bar{0}) e^{ip_4 \cdot X}(z, \bar{z}) \right\rangle \\
&= |\Lambda - 1|^{\alpha' p_1 \cdot p_2}\, |\Lambda|^{\alpha' p_1 \cdot p_3}\, |\Lambda - z|^{\alpha' p_1 \cdot p_4}\, |1 - z|^{\alpha' p_2 \cdot p_4}\, |z|^{\alpha' p_3 \cdot p_4} \delta(P) \\
&\sim |\Lambda|^{\alpha' p_1 \cdot (p_2 + p_3 + p_4)} |1 - z|^{\alpha' p_2 \cdot p_4}\, |z|^{\alpha' p_3 \cdot p_4} \delta(P) \\
&= |\Lambda|^{-\alpha' p_1 \cdot p_1} |1 - z|^{\alpha' p_2 \cdot p_4}\, |z|^{\alpha' p_3 \cdot p_4} = |\Lambda|^{-4} |1 - z|^{\alpha' p_2 \cdot p_4}\, |z|^{\alpha' p_3 \cdot p_4} \delta(P), \tag{5.37}
\end{aligned}$$

where we have used momentum conservation $P \equiv p_1 + p_2 + p_3 + p_3 = 0$ and the mass-shell condition $\alpha' p_i^2 = 4$. Putting together matter and ghost, the $\Lambda$-dependence cancels and we are left with the total amplitude

$$\begin{aligned}
\mathcal{A}(p_1, p_2, p_3, p_4) &= g_s^2 \prod_{i=1}^4 t_i(p_i) \delta(P) \int d^2 z\, |1 - z|^{\alpha' p_2 \cdot p_4}\, |z|^{\alpha' p_3 \cdot p_4} \\
&= g_s^2 \prod_{i=1}^4 t_i(p_i) \delta(P) \int d^2 z\, |1 - z|^{-\alpha' \frac{t}{2} - 4}\, |z|^{-\alpha' \frac{s}{2} - 4}, \tag{5.38}
\end{aligned}$$

where we have defined the Mandelstam invariants

$$s \equiv -(p_1 + p_2)^2 = -(p_3 + p_4)^2, \tag{5.39}$$
$$t \equiv -(p_1 + p_3)^2 = -(p_2 + p_4)^2, \tag{5.40}$$
$$u \equiv -(p_1 + p_4)^2 = -(p_3 + p_2)^2, \tag{5.41}$$

which in the present case obey

$$s + t + u = 4 \times \left( -\frac{4}{\alpha'} \right). \tag{5.42}$$

We can now use the relevant integration formula[26]

$$\int d^2 z\, |z|^{2(a-1)} |1 - z|^{2(b-1)} = 2\pi \frac{\Gamma(a)\Gamma(b)\Gamma(1 - a - b)}{\Gamma(1 - a)\Gamma(1 - b)\Gamma(a + b)}, \tag{5.43}$$

to finally write the *Shapiro-Virasoro* amplitude

$$\mathcal{A}(p_1, p_2, p_3, p_4) = 2\pi g_s^2 \prod_{i=1}^4 t_i(p_i) \delta(P) \frac{\Gamma\left( -\frac{\alpha' s}{4} - 1 \right)\Gamma\left( -\frac{\alpha' t}{4} - 1 \right)\Gamma\left( -\frac{\alpha' u}{4} - 1 \right)}{\Gamma\left( \frac{\alpha' s}{4} + 2 \right)\Gamma\left( \frac{\alpha' t}{4} + 2 \right)\Gamma\left( \frac{\alpha' u}{4} + 2 \right)}. \tag{5.44}$$

This is an example of an amplitude exhibiting *channel duality*, which means invariance under exchange $s \leftrightarrow t \leftrightarrow u$. In a usual field theory this would be achieved only after explicitly adding the three channel contributions (just think of a scalar $\phi^3$ theory, for example). Here instead the amplitude already contains the three channels, without having to explicitly add them, as we would do with QFT Feynman diagrams. This should not be a surprise, this is indeed a *string* amplitude. For generic values of $t$ (and hence of $u$) we can have a look at the $s$-poles of the amplitude. This should give us an indication of the mass of the particles

---

[26]See, for example, Tong's lecture notes for a detailed proof.

that are exchanged in the $s$-channel. The analysis is easy: we have just to look at the poles of $\Gamma\left(-\frac{\alpha's}{4}-1\right)$ which, as it is well-known, appear when the argument of the $\Gamma$ is a non positive integer. This happens precisely when

$$\alpha's = \alpha'm^2 = 4(n-1), \quad n = 0, 1, 2, 3, \dots, \tag{5.45}$$

which is in fact the full mass spectrum of the closed bosonic string! This is why the dual amplitude cannot come from a usual field theory: it describes the scattering of four scalar (tachyonic) particles that interact by exchanging an infinite tower of massive particles. This amplitude could only be reproduced by a field theory with an *infinite* number of fields!

## 5.2 Amplitudes involving open strings

Adding open strings in the game means that the worldsheet is a now a manifold with a boundary and the Gauss-Bonnet theorem is modified to

$$\lambda\chi = \lambda\left(\frac{1}{4\pi}\int_M d^2z\,\sqrt{h}R^{(2)} + \frac{1}{2\pi}\int_{\partial M} ds\,k\right), \tag{5.46}$$

where $k$ is the geodesic curvature of the boundary $\partial M$, defined as

$$k = \frac{1}{2}t^\alpha t^\beta \nabla_\alpha n_\beta, \tag{5.47}$$

where $t$ and $n$ are the tangent and normal (pointing outwards the surface) unit vector to $\partial M$ respectively and $\nabla$ is the covariant derivative. As an example we can consider a disk with flat euclidean metric, whose boundary has radius $R$ and it is parameterized as

$$\gamma(\theta) = R(\cos\theta, \sin\theta), \quad t(\theta) = (-\sin\theta, \cos\theta), \quad n(\theta) = (\cos\theta, \sin\theta), \tag{5.48}$$

so that

$$k = \frac{1}{R}. \tag{5.49}$$

> **Exercise 5.2.1**
>
> Prove eq. (5.49).

Using this we can calculate the Euler characteristic of the disk as

$$\chi = \frac{1}{2\pi}\int_{\partial M} ds\,k = \frac{1}{2\pi}\int_{\partial M}\frac{ds}{R} = \int_0^{2\pi}\frac{d\theta}{2\pi} = 1, \tag{5.50}$$

since in this case the bulk term in $\chi$ does not contribute. If we do the same computation for an *annulus* we find this time two cancelling contributions (because the normal vectors of the two boundaries of the annulus are oppositely oriented). In general for a worldsheet of genus $g$ and $b$ boundaries we have

$$\chi = 2(1-g) - b. \tag{5.51}$$

Moreover, if we also consider $n_o$ open and $n_c$ closed string insertions we have an additional change in the value of $\chi$ which in the most general case becomes

$$\chi = 2(1-g) - b - n_c - \frac{n_o}{2}. \tag{5.52}$$

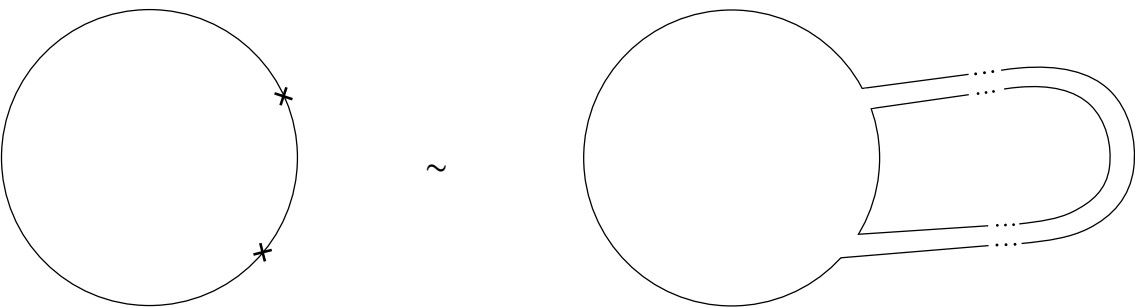

Figure 5.8: A disk with two punctures on the boundary is homeomorphic to a disk with one more (inner) boundary.

The $-1/2$ factor in front of the open string number $n_o$ comes from a similar reasoning as the one followed to fix to $-1$ the coefficient in front $n_c$. Let us indeed suppose to add to a boundary two new boundary punctures. These two can be thought as two infinite spikes coming out from the boundary and fusing at infinity creating a new inner boundary (see fig. 5.8).

This means that adding 2 boundary punctures changes the value of $\chi$ as the insertion of a new boundary in the worldsheet, namely $\Delta\chi = -1$. Because two punctures change the value of $\chi$ by $-1$, a single puncture will contribute by a factor of $-1/2$.

Putting all of the above informations together we can then write a complete open-closed string scattering amplitude:

$$\mathcal{A}(n_c, n_o) = g_s^{n_c + \frac{n_o}{2}} \sum_{n=0}^{\infty} g_s^{2(g-1)} \sum_{b=1}^{\infty} g_s^{b} \langle\!\langle \mathcal{V}_1^c \cdots \mathcal{V}_{n_c}^c \mathcal{V}_1^o \cdots \mathcal{V}_{n_o}^o \rangle\!\rangle_{g,b}, \tag{5.53}$$

where as usual $g_s = g_{\text{closed}} = e^{\lambda}$. The summation over topologies must now also take care of the presence of boundaries, so it also runs over $b$. We can also notice that for every open string in the amplitude there is an extra factor of $g_s^{1/2}$ which means that the open string coupling $g_{\text{open}}$ can be seen as the square root of the closed string coupling:

$$g_{\text{open}} = \sqrt{g_{\text{closed}}}. \tag{5.54}$$

The term $\langle\!\langle \mathcal{V}_1^c \cdots \mathcal{V}_{n_c}^c \mathcal{V}_1^o \cdots \mathcal{V}_{n_o}^o \rangle\!\rangle_{g,b}$ denotes a correlation function of gauge invariant operators corresponding to closed and open string punctures, computed on a $2d$ manifold with genus $g$ and $b$ boundaries.

The general form of an open-closed amplitude on a Riemann surface with $b$ boundaries is combinatorially rather cumbersome: indeed the path integral prescribes to sum over all possible ways of distributing $n_o$ open string punctures on the available $b$ boundaries and, on every boundary, to integrate over all possible positions on the boundary, paying attention to the path-ordering prescription. Every boundary will then give rise to a trace in the Chan-Paton matrices associated to the open string insertions. In this introductory lectures we will only be interested in tree-level processes, associated with a disk with boundary and bulk punctures which is conformally equivalent to the (punctured) UHP. We will thus consider

$$\langle\!\langle \mathcal{V}_1^c \cdots \mathcal{V}_{n_c}^c \mathcal{V}_1^o \cdots \mathcal{V}_{n_o}^o \rangle\!\rangle_{g=0,b=1} = \int \frac{\mathcal{D}X\mathcal{D}h}{\text{Vol}(\mathcal{G})} e^{-S_{\text{Pol}}[X,h]}$$

$$\times \int_{\mathbb{R}} dx_1 \cdots \int_{\mathbb{R}} dx_{n_o} \, \text{Tr}\, \mathcal{P}[\mathcal{V}_1^o(x_1) \cdots \mathcal{V}_{n_o}^o(x_{n_o})]$$

$$\times \int_{\text{UHP}} d^2 z_1 \cdots \int_{\text{UHP}} d^2 z_{n_c} \mathcal{V}_1^c(z_1, \bar{z}_1) \cdots \mathcal{V}_{n_c}^c(z_{n_c}, \bar{z}_{n_c}). \tag{5.55}$$

As already discussed, because the boundary is 1-dimensional we need to take care of the operator ordering. That is why we have introduced the path ordering operator $\mathcal{P}$ defined as

$$\mathcal{P}\big[\cdots \mathcal{V}_i(x_i)\cdots \mathcal{V}_j(x_j)\cdots\big] = \begin{cases} \big[\cdots \mathcal{V}_i(x_i)\cdots \mathcal{V}_j(x_j)\cdots\big], & x_i > x_j, \\ (-1)^\epsilon \big[\cdots \mathcal{V}_j(x_j)\cdots \mathcal{V}_i(x_i)\cdots\big], & x_j < x_i, \end{cases} \tag{5.56}$$

where the $(-1)^\epsilon$ takes care of the grassmannality of the operators.

## 5.3 Tree level open string amplitudes

Let us now gauge fix Diff$\times$ Weyl and focus on a pure open string scattering at tree level

$$\langle\!\langle \mathcal{V}_1^o \cdots \mathcal{V}_{n_o}^o \rangle\!\rangle_{g=0,b=1} = \text{Tr} \int \mathcal{D}X\mathcal{D}c\mathcal{D}b \int_{\mathbb{R}} dx_1\cdots dx_{n_o}\, \mathcal{P}\big[\mathcal{V}_1^o(x_1)\cdots \mathcal{V}_{n_o}^o(x_{n_o})\big] e^{-S[X,b,c]}$$

$$= \text{Tr} \int_{\mathbb{R}} dx_1\cdots dx_{n_o}\, \mathcal{P}\big\langle \mathcal{V}_1^o(x_1)\cdots \mathcal{V}_{n_o}^o(x_{n_o})\big\rangle_{\text{UHP}}, \tag{5.57}$$

where in the last line we have explicitly written the matter+ghost path integral as a BCFT correlator. Just as in the closed string case this expression is formally $0 \times \infty$, the zero coming from the unsaturated $c$ zero modes and the infinity coming from the overcounting proportional to the $SL(2,\mathbb{R})$ volume. To gauge fix this redundancy we need again to introduce three non-integrated operators with an explicit dependence on the boundary

$$\int dx_i \mathcal{V}_i^o(x_i) \to c\mathcal{V}_i^o(x_i), \quad i = 1,2,3. \tag{5.58}$$

However, when fixing these three open strings we have to sum over the two configurations which are not cyclically equivalent, for example $(1,2,3) + (1,3,2)$.[27] Thus the tree level amplitude takes the form

$$\langle\!\langle \mathcal{V}_1^o \cdots \mathcal{V}_{n_o}^o \rangle\!\rangle_{g=0,b=1} = \text{Tr} \int_{\mathbb{R}} dx_4\cdots dx_{n_o}\, \mathcal{P}\langle c\mathcal{V}_1(x_1)c\mathcal{V}_2(x_2)c\mathcal{V}_3(x_3)\mathcal{V}(x_4)\cdots \mathcal{V}(x_{n_o})\rangle$$

$$+ \text{Tr} \int_{\mathbb{R}} dx_4\cdots dx_{n_o}\, \mathcal{P}\langle c\mathcal{V}_1(x_1)c\mathcal{V}_3(x_2)c\mathcal{V}_2(x_3)\mathcal{V}(x_4)\cdots \mathcal{V}(x_{n_o})\rangle. \tag{5.59}$$

Clearly, just as in the closed string case, the amplitude does not depend on the choice of $(x_1, x_2, x_3)$. Moreover it does also not depend on which vertex operators we decide to fix.

### 5.3.1 Three tachyons amplitude

At first we consider the simplest example possible: the scattering of three open string tachyons in a $D(25)$-brane background (only Neumann condition) at tree level ($g = 0$, $b = 1$). The three tachyons will have momenta $P_1, P_2, P_3$ that by construction must satisfy

$$P_1 + P_2 + P_3 = 0, \quad \alpha' P_i^2 = 1, \quad i = 1,\ldots,3, \tag{5.60}$$

and will be created by the simplest matter vertex operator, namely

$$\mathcal{V}_i(x) = t(P_i) :e^{iP_i \cdot X}:(x). \tag{5.61}$$

---

[27]In the gauge unfixed form, with only integrated open string vertex operators, this sum would have been implicitly included in the triple integration $\int_{-\infty}^{\infty} dx_1\,dx_2\,dx_3$.

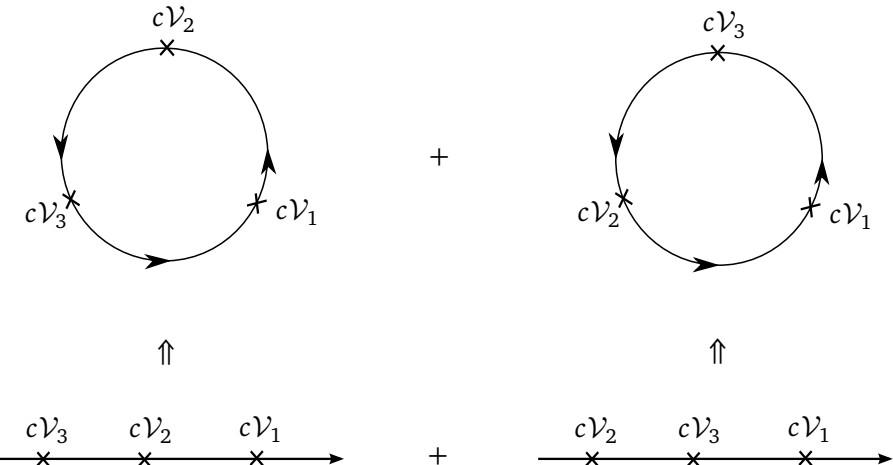

Figure 5.9: The two possible boundary orderings for three operators.

The amplitude then takes the form

$$\mathcal{A}(1,2,3) = g_s^{\frac{1}{2}}\Big(\langle c\mathcal{V}_1(x_1)c\mathcal{V}_2(x_2)c\mathcal{V}_3(x_3)\rangle + \langle c\mathcal{V}_1(x_1)c\mathcal{V}_3(x_2)c\mathcal{V}_2(x_3)\rangle\Big),\qquad(5.62)$$

where we have summed over the two possible non-equivalent orderings on the boundary of the disk (see fig. 5.9).

The computation is easy because we know that

$$\langle c(x_1)c(x_2)c(x_3)\rangle = x_{12}x_{13}x_{23},\qquad(5.63)$$

$$\langle e^{iP_1\cdot X}(x_1)e^{iP_2\cdot X}(x_2)e^{iP_3\cdot X}(x_3)\rangle = x_{12}^{2\alpha' P_1\cdot P_2}x_{13}^{2\alpha' P_1\cdot P_3}x_{23}^{2\alpha' P_2\cdot P_3},\qquad(5.64)$$

where we used the definition $x_{ij} = x_i - x_j$. Then squaring $P_1 + P_2 + P_3 = 0$ we get

$$P_1^2 + P_2^2 + 2P_1\cdot P_2 = P_3^2 \implies P_1\cdot P_2 = -\frac{1}{2\alpha'} = P_i\cdot P_j,\quad\text{for}\quad i\neq j,\qquad(5.65)$$

so that the three matter vertices produce a contribution equal to $(x_{12}x_{13}x_{23})^{-1}$ that perfectly cancels with the ghost contribution. This gives a result which is independent of the choice of the three points on the boundary of the disk:

$$\mathcal{A}(1,2,3) = 2g_s^{\frac{1}{2}}\,t(P_1)t(P_2)t(P_3)\delta(P_1+P_2+P_3),\qquad(5.66)$$

where the other ordering (giving the same result) has been included.

### 5.3.2 Veneziano amplitude

A more complex case is represented by the tree-level ($g=0, b=1$) four-open-string-tachyon amplitude. This amplitude is also known as the *Veneziano Amplitude*. Explicitly we write

$$\mathcal{A}(1,\dots,4) = g_s\int_{-\infty}^{+\infty} dx\,\mathcal{P}\langle c\mathcal{V}_1(x_1)c\mathcal{V}_2(x_2)c\mathcal{V}_3(x_3)\mathcal{V}_4(x)\rangle$$
$$+ g_s\int_{-\infty}^{+\infty} dx\,\mathcal{P}\langle c\mathcal{V}_1(x_1)c\mathcal{V}_3(x_3)c\mathcal{V}_2(x_2)\mathcal{V}_4(x)\rangle.\qquad(5.67)$$

We start to compute the first of the two integrals. The second will just give the same result in this simple case. Using the $SL(2,\mathbb{R})$ invariance to fix arbitrarily $x_1 = \Lambda \to \infty$, $x_2 = 1$, $x_3 = 0$ the correlators give the following results:

$$\langle c(\Lambda)c(1)c(0) \rangle = \Lambda(\Lambda - 1), \tag{5.68}$$

$$\langle e^{iP_1 \cdot X}(\Lambda) e^{iP_2 \cdot X}(1) e^{iP_3 \cdot X}(0) e^{iP_4 \cdot X}(x) \rangle = (\Lambda - 1)^{2\alpha' P_1 \cdot P_2} \Lambda^{2\alpha' P_1 \cdot P_3} (\Lambda - x)^{2\alpha' P_1 \cdot P_4}$$
$$\times |x|^{2\alpha' P_3 \cdot P_4} |1 - x|^{2\alpha' P_2 \cdot P_4}. \tag{5.69}$$

Notice that the absolute value is a clever shortcut to right a result which is valid for all possible positions of $x$ with respect to $0, 1$. Since $\Lambda \to \infty$, we take $x < \Lambda$ without loosing generality. When $\Lambda \to \infty$ the first three terms of the matter correlator reduce to

$$\Lambda^{2\alpha'(P_2 + P_3 + P_4) \cdot P_1} = \Lambda^{-2\alpha' P_1^2} = \Lambda^{-2}, \tag{5.70}$$

that we have simplified using the on-shell condition $\alpha' P_i^2 = 1$, $i = 1, \ldots, 4$ and the momentum conservation $P_1 + P_2 + P_3 + P_4 = 0$. Putting everything together we get

$$\langle c\mathcal{V}_1(\Lambda) c\mathcal{V}_2(1) c\mathcal{V}_3(0)\mathcal{V}_4(x) \rangle \Big|_{\Lambda \to \infty} = \frac{\Lambda(\Lambda - 1)}{\Lambda^2} \Big|_{\Lambda \to \infty} |x|^{2\alpha' P_3 \cdot P_4} |1 - x|^{2\alpha' P_2 \cdot P_4}$$
$$= |x|^{2\alpha' P_3 \cdot P_4} |1 - x|^{2\alpha' P_2 \cdot P_4}. \tag{5.71}$$

Now we have to integrate this form over the real line, which splits into three different parts:

$$\int_{-\infty}^{+\infty} dx |x|^{2\alpha' P_3 \cdot P_4} |1 - x|^{2\alpha' P_2 \cdot P_4} = I_1 + I_2 + I_3, \tag{5.72}$$

with

$$I_1 = \int_0^1 dx \, x^{2\alpha' P_3 \cdot P_4} (1 - x)^{2\alpha' P_2 \cdot P_4}, \tag{5.73a}$$

$$I_2 = \int_1^\infty dx \, x^{2\alpha' P_3 \cdot P_4} (x - 1)^{2\alpha' P_2 \cdot P_4}, \tag{5.73b}$$

$$I_3 = \int_{-\infty}^0 dx \, (-x)^{2\alpha' P_3 \cdot P_4} (1 - x)^{2\alpha' P_2 \cdot P_4}. \tag{5.73c}$$

The second and third integrals can be nicely recast in a form similar to $I_1$ by changing the integration variable and using momentum conservation plus on-shell condition. For $I_2$ we choose $x = 1/y$, $dx = -dy/y^2$ so that

$$I_2 = -\int_1^0 \frac{dy}{y^2} y^{-2\alpha' P_3 \cdot P_4} \left( \frac{1 - y}{y} \right)^{2\alpha' P_2 \cdot P_4}$$
$$= \int_0^1 dy \, y^{-2 - 2\alpha'(P_2 + P_3) \cdot P_4} (1 - y)^{2\alpha' P_2 \cdot P_4}$$
$$= \int_0^1 dx \, x^{2\alpha' P_1 \cdot P_4} (1 - y)^{2\alpha' P_2 \cdot P_4}. \tag{5.74}$$

For $I_3$ on the other hand we define $x = y/(y-1)$, $dx = \frac{dy}{(1-y)^2}$ so that

$$
\begin{aligned}
I_3 &= -\int_1^0 \frac{dy}{(1-y)^2} \left(\frac{y}{1-y}\right)^{2\alpha' P_3 \cdot P_4} (1-y)^{-2\alpha' P_2 \cdot P_4} \\
&= \int_0^1 dy \, y^{2\alpha' P_3 \cdot P_4} (1-y)^{-2-2\alpha' P_4 \cdot (P_3 + P_2)} \\
&= \int_0^1 dx \, x^{2\alpha' P_3 \cdot P_4} (1-x)^{2\alpha' P_1 \cdot P_4} \, .
\end{aligned}
\tag{5.75}
$$

If we consider the Mandelstam variables:

$$
s = -(P_1 + P_2)^2 \,, \qquad t = -(P_1 + P_3)^2 \,, \qquad u = -(P_1 + P_4)^2 \,,
\tag{5.76}
$$

with

$$
s + t + u = 4 \times \left(-\frac{1}{\alpha'}\right), \qquad \alpha' P_i \cdot P_j = -2 - \alpha' s_{ij} \,,
\tag{5.77}
$$

where

$$
s_{12} = s \,, \qquad s_{13} = t \,, \qquad s_{14} = u \,,
\tag{5.78}
$$

it is obvious that the final amplitude can then be written as

$$
\mathcal{A}(1,\ldots,4) = 2 g_o^2 \delta(P_1 + P_2 + P_3 + P_4) \left[ I(s,t) + I(t,u) + I(u,s) \right] ,
\tag{5.79}
$$

where we have introduced the open string coupling $g_o^2 = g_s$ and the integral function

$$
I(a,b) = \int_0^1 dx \, x^{-2-\alpha' a} (1-x)^{-2-\alpha' b} = B(-1-\alpha' a, -1-\alpha' b),
\tag{5.80}
$$

with $B$ the Euler beta function. The properties of the Beta function are now important to understand the phenomenology.

1. The property $B(x,y) = B(y,x)$ is another example of *channel duality* which is a distinctive feature of strings amplitudes and which is rather unusual for a QFT. In a typical QFT with a finite number of interacting fields one has to add to the $s$ channel contribution explicit contributions from the $t$ and $u$ channels. This is not the case in string theory: the channel duality says that, for example, in the $s$-channel interpretation of $I(s,t)$ we don't have to add the $t$ channel since it is already included in the infinite sum of the $s$ channel poles. Channel duality was postulated in the prehistory of string theory as a typical property of interactions of hadron resonances organizing themselves into Regge trajectories. In 1968 Veneziano smartly wrote down his amplitude as the simplest elementary function having explicit channel duality, without having an underlying understanding of what kind of theory could give rise to this amplitude. Then a lot of fun came by analyzing the pole structure of that amplitude and this resulted in the birth of String Theory.

2. Because the scattering is at tree level, we can represent its $s$-channel with the diagram in fig. 5.10. By looking at the $s$ simple poles in the amplitudes we can read-off the mass of the exchanged particles inside the inner propagator. We can then study the singularities of $I(s,t)$ writing it in terms of Gamma functions:

$$
I(s,t) = \frac{\Gamma(-\alpha' s - 1)\Gamma(-\alpha' t - 1)}{\Gamma(-\alpha'(s+t) - 2)} \, .
\tag{5.81}
$$

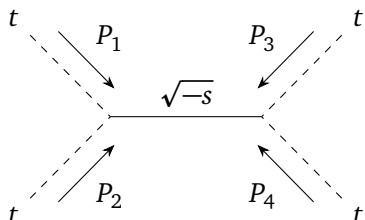

Figure 5.10: The $s$-channel diagram of the Veneziano amplitude.

Since near the poles of the $\Gamma$ we have

$$\Gamma(z) = \frac{(-1)^n}{n!}\frac{1}{z+n}, \quad \text{for } z \sim -n \in -\mathbb{N}, \tag{5.82}$$

it is clear that $I(s,t)$ has poles for $\alpha's = n-1$ exaclty in correspondence of particles of mass $m^2 = \frac{1}{\alpha'}(n-1)$ which is precisely the tower of masses of the open string spectrum. This means that the amplitude is correctly describing the interaction of the external states with an infinite tower of massive particles.

3. We can now expand $I(s,t)$ around the $s$ poles to get

$$
\begin{aligned}
I(s,t) &= \frac{(-1)^n}{n!}\frac{1}{n-1-\alpha's}\frac{\Gamma(-1-\alpha't)}{\Gamma(-1-n-\alpha't)} \\
&= \frac{1}{n!}\frac{1}{n-1-\alpha's}(\alpha't+1)\dots(\alpha't+n+1) \\
&\overset{t\to\infty}{\sim} \frac{(\alpha't)^n}{s+m_n^2},
\end{aligned}
\tag{5.83}
$$

with $\alpha'm_n = (n-1)$. Notice that in the numerator we get a contribution $t^n$ which gives us an information over the spin of the exchanged particle. Indeed for a scattering between scalars due to the exchange of a particle of spin $J$ the amplitude is of the form

$$\mathcal{A} \sim \frac{t^J}{s+m^2}, \quad s \sim -m^2, \tag{5.84}$$

because a typical cubic vertex between two scalars and a spin $J$ field (represented by a polarization tensor with $J$ fully symmetric Lorentz indices) necessarily brings a power $\sim p^J$ from the $J$ derivatives which have to contract with the polarization in a coupling that, without further specification, will have a rough form $\sim \psi^{\mu_1\cdots\mu_J}[\phi \overleftrightarrow{\partial}_{\mu_1\cdots\mu_J}\phi]$. Therefore a quartic amplitude obtained by 2 cubic vertices connected by a propagator will necessarily have a numerator proportional to $\sim p^{2J} \sim t^J$ (for large $t$ and close to the pole $s \sim -m^2$). This is clearly consistent with what we know from the string spectrum, where at level $n$ we find particles with spin $J = n$ and lower. The particles with maximal spin $J = n$ inside the string spectrum formula gives rise to the mass-spin relation

$$\alpha'm^2 = J-1, \tag{5.85}$$

which is the equation of the first *Regge trajectory*.

At the time of this discovery this was one of the reasons to believe that string theory could describe hadron interactions. Today we know that the fundamental theory behind hadron resonances is instead QCD, a more traditional (but not so traditional at that time!) Yang-Mills theory. At low energy QCD becomes strongly coupled and confines (a physical phenomenon

whose theoretical mechanism is still rather mysterious) and quarks get bound with antiquarks by flux tubes which can be well-described by an effective string. Therefore a Yang-Mills Theory like QCD can have, in its strong-coupling phase, a description in terms of effective strings! This is yet another astonishing fact about strings which is at the hearth of the celebrated $AdS/CFT$ correspondence.

### 5.3.3 Three gluons scattering

Let us now consider a background with $N$ coincident $D(25)$ branes and a three-gluons amplitude. The matter boundary vertex operator is

$$\mathcal{V}(x) = \mathbb{A}_\mu(P) :j^\mu e^{iP\cdot X}: (x), \quad \text{with} \quad P^2 = 0, \tag{5.86}$$

and $\mathbb{A}_\mu(P)$ an $N \times N$ matrix such that $P_\mu \mathbb{A}^\mu = 0$. Because we are dealing with a tree level ($g = 0, b = 1$) three point function the amplitude will be of the same form of (5.62), with an overall coupling given by $g_c^{2g-2+b-n_c/2} = g_c^{-2+1-3/2} = g_c^{1/2} = g_o$:

$$\mathcal{A} = g_o \underbrace{\text{Tr}\langle c\mathcal{V}_1(x_1)c\mathcal{V}_2(x_2)c\mathcal{V}_3(x_3)\rangle}_{\mathcal{A}_{123}} + g_o \underbrace{\text{Tr}\langle c\mathcal{V}_1(x_1)c\mathcal{V}_3(x_2)c\mathcal{V}_2(x_3)\rangle}_{\mathcal{A}_{132}}, \tag{5.87}$$

where the trace is now done on the $U(N)$ indices of $\mathbb{A}_\mu(P)$. Focusing now on the first term and choosing again $x_1 = \Lambda \to \infty$, $x_2 = 1$, $x_3 = 0$ we are left with the following correlator:

$$\begin{aligned}
\mathcal{A}_{123} &= \text{Tr}\langle c\mathbb{A}_\mu(P_1) :j^\mu e^{iP_1\cdot X}: (\Lambda) c\mathbb{A}_\nu(P_2) :j^\nu e^{iP_2\cdot X}: (1) c\mathbb{A}_\rho(P_3) :j^\rho e^{iP_3\cdot X}: (0)\rangle \\
&= \Lambda(\Lambda-1)\text{Tr}\left[\mathbb{A}_\mu(P_1)\mathbb{A}_\nu(P_2)\mathbb{A}_\rho(P_3)\right]\langle :j^\mu e^{iP_1\cdot X}: (\Lambda) :j^\nu e^{iP_2\cdot X}: (1) :j^\rho e^{iP_3\cdot X}: (0)\rangle. 
\end{aligned} \tag{5.88}$$

Now the computation of the matter correlator results to be a little bit more complicated because we have many ways to contract the operators together. Using the properties of the currents and the exponential after a tedious but straightforward calculation we are left with

$$\mathcal{A}_{123} = \text{Tr}\left[\mathbb{A}_\mu^{(1)}\mathbb{A}_\nu^{(2)}\mathbb{A}_\rho^{(3)}\right]\left(P_1^\rho \eta^{\mu\nu} + P_2^\mu \eta^{\rho\nu} + P_3^\nu \eta^{\rho\mu} + 2\alpha' P_1^\rho P_2^\mu P_3^\nu\right)\delta(P), \tag{5.89a}$$

$$\mathcal{A}_{132} = \text{Tr}\left[\mathbb{A}_\mu^{(1)}\mathbb{A}_\rho^{(3)}\mathbb{A}_\nu^{(2)}\right]\underbrace{\left(P_1^\nu \eta^{\mu\rho} + P_3^\mu \eta^{\nu\rho} + P_2^\rho \eta^{\mu\nu} + 2\alpha' P_1^\nu P_3^\mu P_2^\rho\right)}_{t_{132}^{\mu\rho\nu}}\delta(P), \tag{5.89b}$$

where $P = P_1 + P_2 + P_3$, $\mathbb{A}^{(i)} = \mathbb{A}^{(i)}(P_i)$. Now we can put them together and rewrite the full amplitude as

$$\mathcal{A} = g_o\delta(P)\text{Tr}\left[\mathbb{A}_\mu^{(1)}\mathbb{A}_\rho^{(3)}\mathbb{A}_\nu^{(2)}\right](t_{123}^{\mu\nu\rho} + t_{132}^{\mu\rho\nu}) + g_o\delta(P)t_{123}^{\mu\nu\rho}\text{Tr}\left[\mathbb{A}_\mu^{(1)}[\mathbb{A}_\nu^{(2)},\mathbb{A}_\rho^{(3)}]\right], \tag{5.90}$$

where we simply added and subtracted $\delta(P)\text{Tr}\left[\mathbb{A}_\mu^{(1)}\mathbb{A}_\rho^{(3)}\mathbb{A}_\nu^{(2)}\right]t_{123}^{\mu\nu\rho}$. Then we can explicitly compute

$$t_{123}^{\mu\nu\rho} + t_{132}^{\mu\rho\nu} = -P_1^\mu\left(\eta^{\nu\rho} - 2\alpha' P_3^\nu P_2^\rho\right) - P_2^\nu\left(\eta^{\mu\rho} - 2\alpha' P_3^\mu P_2^\rho\right) - P_3^\rho\left(\eta^{\mu\nu} - 2\alpha' P_2^\mu P_3^\nu\right) = 0, \tag{5.91}$$

because $P_i^\mu \mathbb{A}_\mu(P_i)$. So only the commutator term survives and the amplitude takes the form

$$\mathcal{A} = g_o\,\delta(P)t_{123}^{\mu\nu\rho}\text{Tr}\left[\mathbb{A}_\mu^{(1)}[\mathbb{A}_\nu^{(2)},\mathbb{A}_\rho^{(3)}]\right]. \tag{5.92}$$

This result is very important. Just looking at the spectrum on $N$ coincident D-branes we were finding $N^2$ gauge bosons, but we didn't have any information about their interactions. Now we know that they interact as in a $U(N)$ Yang-Mills theory! In particular for $N = 1$ we notice that the amplitude goes to zero because $[\mathbb{A}^{(1)}, \mathbb{A}^{(2)}] = 0$. Looking at $\mathcal{A}$ we can try to write down

an effective action that reproduces it. Knowing the free equation of motion derived from the spectrum and looking at the terms in $\mathcal{A}$ which are linear in the momenta, we can write the effective lagrangian that reproduce them as

$$\mathcal{L}_{\text{eff}} \supset -\frac{1}{4}\operatorname{Tr}\big[\partial_\mu A_\nu - \partial_\nu A_\mu\big]^2 - \frac{1}{2}g_o\operatorname{Tr}\big[\partial_\mu A_\nu[A^\mu, A^\nu]\big], \tag{5.93}$$

which are exactly two of the three terms appearing in the Yang-Mills Lagrangian. To see the remaining term, the commutator squared $\operatorname{Tr}\big[[A_\mu, A_\nu][A^\mu, A^\nu]\big]$, we should compute the 4-gluon scattering. This can be done, although not in these introductory lectures, and indeed, after subtracting the contribution from two previously found cubic vertices connected by the gluon propagator, one can reproduce the full YM lagrangian

$$\mathcal{L}_{\text{eff}} \supset -\frac{1}{4}\operatorname{Tr}\big[F_{\mu\nu}F^{\mu\nu}\big], \tag{5.94}$$

where $F_{\mu\nu} = \partial_\mu A_\nu - \partial_\nu A_\mu + ig_{\text{YM}}[A_\mu, A_\nu]$ and where we identify $g_{\text{YM}} = g_o$.

On top of this, inside $\mathcal{A}$ we also have a new term $\propto \alpha' P_1 P_2 P_3$ which is reproduced by a term in the action of the form $\alpha' \operatorname{Tr}\big[\partial_{[\mu}A_{\nu]}\partial^{[\nu}A^{\rho]}\partial_{[\rho}A_{\sigma]}\big]\eta^{\sigma\mu}$. This term is clearly non-renormalizable and IR irrelevant and is an example of an $\alpha'$-correction. If we compute the string amplitudes up to 6-points, we would explicitly verify that this term contributes to a completely new non-renormalizable gauge invariant term in the effective Yang-Mills Lagrangian

$$\mathcal{L}_{\text{eff}} \supset -\frac{1}{4}\operatorname{Tr}\big[F_{\mu\nu}F^{\mu\nu} + \alpha' F_\mu{}^\nu F_\nu{}^\rho F_\rho{}^\mu\big] + O(\alpha'^2). \tag{5.95}$$

This is an example of how string theory modifies well-known QFT's like Yang-Mills by adding irrelevant deformations controlled by the string scale. This is how the effective action is trying to tell us that what we are studying is not just a theory of gluons, but it is in fact a (open in this case) *string theory*.

## 5.4  Strings in curved background

In this section we would like to understand how the full Einstein equations are predicted by String Theory. To do so, let us consider the *non-linear sigma model* which we get by taking the Polyakov action and substituting the flat metric $\eta_{\mu\nu}$ with a generic metric $G_{\mu\nu}(X)$ on the target space $M$:

$$S[X] = \frac{1}{2\pi\alpha'}\int d^2z\,\partial X^\mu\bar\partial X^\nu G_{\mu\nu}(X). \tag{5.96}$$

If we suppose

$$G_{\mu\nu}(X) = \eta_{\mu\nu} + h_{\mu\nu}(X), \tag{5.97}$$

with infinitesimal $h_{\mu\nu}$, then

$$\begin{aligned} S_{\eta+h} &= \frac{1}{2\pi\alpha'}\int d^2z\,\partial X^\mu\bar\partial X^\nu\eta_{\mu\nu} + \frac{1}{2\pi\alpha'}\int d^2z\,\partial X^\mu\bar\partial X^\nu h_{\mu\nu}(X)\\ &= S_\eta + \frac{1}{2\pi\alpha'}\int d^2z\,V_h(z,\bar z), \end{aligned} \tag{5.98}$$

where we defined the vertex operator of the graviton as

$$\mathcal{V}_h(z,\bar z) = \int dp\,h_{\mu\nu}(p)\partial X^\mu\bar\partial X^\nu e^{ip\cdot X}(z,\bar z), \tag{5.99}$$

using the Fourier transform

$$h_{\mu\nu}(X) = \int dp\, e^{ip\cdot X} h_{\mu\nu}(p)\,. \tag{5.100}$$

Therefore an infinitesimal deformation of the background metric is obtained by deforming the original world-sheet action with the integrated vertex operator of the graviton.

Classically the insertion of $\mathcal{V}_h$ satisfies conformal invariance. From the quantum point of view, however, $\mathcal{V}_h$ has to be a primary of weight $(1,1)$, which is true if (see eq. (4.155))

$$\begin{cases} P^2 = 0\,, \\ P_\mu h^{\mu\nu} = h^{\mu\nu} P_\nu = 0\,, \end{cases} \tag{5.101}$$

which imply the linearized Einstein equations. Hence, when we slightly modify the flat metric, the theory preserves conformal invariance (and thus remains consistent as a string theory) if the fluctuation satisfies the linearized Einstein equations.

To go further, the full path integral is deformed as follows:

$$Z = \int \mathcal{D}X\, e^{-S[X] - \int d^2z\, \mathcal{V}_h(z,\bar{z})} = \int \mathcal{D}X\, e^{-S[X]} e^{-\int d^2z\, \mathcal{V}_h(z,\bar{z})}\,, \tag{5.102}$$

with

$$e^{-\int d^2z\, \mathcal{V}_h(z,\bar{z})} = 1 - \int d^2z\, \mathcal{V}_h(z,\bar{z}) - \frac{1}{2}\int d^2z'\int d^2z\, \mathcal{V}_h(z,\bar{z})\mathcal{V}_h'(z',\bar{z}') + \dots\,, \tag{5.103}$$

which is an insertion of a sort of coherent state of gravitons. In this approach, however, it is not easy to regulate the collisions of the graviton vertex operators, to end up with renormalized operators. This is in principle possible and it would give non-linear conditions on $h_{\mu\nu}$ and its space-time derivatives to ensure that the renormalized operators remain conformal. If we could perform this *conformal perturbation theory* to every order we would then get the full space-time equations for $h_{\mu\nu}$ and we could compare them with Einstein equation. However we don't know how to do this in generality. Let us thus follow an alternative route and reconsider the initial sigma-model (5.96). If we expand $G_{\mu\nu}$ nearby a $X_0 \in M$ we get

$$G_{\mu\nu}(X) = G_{\mu\nu}(X_0) + G_{\mu\nu,\rho}(X_0)Y^\rho + G_{\mu\nu,\rho\sigma}(X_0)Y^\rho Y^\sigma + \mathcal{O}(Y^3)\,, \tag{5.104}$$

with $Y = X - X_0$. Then the action reads

$$S[Y] = \frac{1}{2\pi\alpha'}\int d^2z\, \left(\partial Y^\mu \bar{\partial} Y^\nu G_{\mu\nu}(X_0) + \partial Y^\mu \bar{\partial} Y^\nu \cdot Y^\rho G_{\mu\nu,\rho}(X_0) + \dots \right)\,, \tag{5.105}$$

where the first contribution of the sum is the kinetic term, while the others are infinitely many interactions with $G_{\mu\nu,\rho},\dots$ serving as coupling constants.

Before the deformations this 2$d$-QFT was conformal. Now we are deforming it with an $\infty$ number of interactions that in the quantum theory will be renormalized. The theory preserves its conformal symmetry if the coupling constants are unchanged under the RG flow (and thus if their $\beta$ functions vanish). Since in string theory conformal symmetry is a gauge symmetry and cannot be broken, we are only interested in the fixed points of the RG flow.

To simplify this problem, let us choose a system of geodesic coordinates, which can be further specialized to Riemann normal coordinates:

$$G_{\mu\nu}(X) = \underbrace{G_{\mu\nu}(X_0)}_{\eta_{\mu\nu}} - \frac{1}{3}R_{\mu\rho\nu\sigma}(X-X_0)^\rho(X-X_0)^\sigma + \mathcal{O}(X^3)\,. \tag{5.106}$$

In a two-dimensional context it is usually convenient to use dimensionless fields, namely $[Y] = \emptyset$. But since $[X] = L$ we take

$$X^\mu = X_0^\mu + \sqrt{\alpha'}Y^\mu. \tag{5.107}$$

Thus

$$G_{\mu\nu}(X) = G_{\mu\nu}(X_0) - \frac{1}{3}\alpha' R_{\mu\rho\nu\sigma}Y^\rho Y^\sigma + \mathcal{O}(X^3). \tag{5.108}$$

Then the actions reads

$$S[Y] = \frac{1}{2\pi\alpha'}\int d^2z\,\alpha'\partial Y^\mu\bar{\partial}Y^\nu\left(\eta_{\mu\nu} - \frac{1}{3}\alpha' R_{\mu\rho\nu\sigma}Y^\rho Y^\sigma + \dots\right). \tag{5.109}$$

This $2d$-QFT has the following Feynman rules:

$$\mu\ \xrightarrow{\ k\ }\ \nu\ \xrightarrow{\text{FT}}\ \frac{\eta_{\mu\nu}}{k^2}, \tag{5.110}$$

$$\xrightarrow{\text{FT}}\ \frac{\alpha'}{3}R_{\mu\rho\nu\sigma}(k_1\cdot k_2), \tag{5.111}$$

where the dot $\cdot$ is the scalar product in 2 dimensions.

In order to do perturbation theory we have to ask for a small coupling constant:

$$\alpha' R \ll 1, \tag{5.112}$$

which means that the spacetime is curved just a little with respect to the string scale, i.e. the string is so small that sees the spacetime as if it was flat.

Let us now evaluate the self energy of the scalar $Y$:

$$\propto (k_1\cdot k_2)\,\delta(k_1+k_2)R_{\mu\rho\nu}{}^\rho\int d^2q\,\frac{1}{q^2}. \tag{5.113}$$

In order to regulate the logarithmic divergence $\int^\infty dq/q$ we can use dimensional regularization:

$$\int d^2q\,\frac{1}{q^2} \to \mu^{-\epsilon}\int\frac{d^{2+\epsilon}q}{q^2} \sim \mu^{-\epsilon}\int^\infty dq\,q^{\epsilon-1} \sim \mu^{-\epsilon}\frac{1}{\epsilon}, \tag{5.114}$$

where we have introduced the renormalization (mass) scale $\mu$ to keep the dimensions of the amplitude fixed.[28] This implies

$$\propto \mu^{-\epsilon}\frac{R_{\mu\nu}}{\epsilon}(k_1\cdot k_2)\delta(k_1+k_2). \tag{5.115}$$

---

[28]We hope the reader won't confuse the renormalization scale $\mu$ with the Lorentz index $^\mu$.

We can now renormalize the action, absorbing the divergence into a counterterm as follows:

$$S_{\text{ren}} = S_{\text{bare}} + \int d^2z \, \partial Y^\mu \bar{\partial} Y^\nu \left(-\frac{\alpha'}{3}\right) \mu^{-\epsilon} \frac{R_{\mu\nu}}{\epsilon} = \int d^2z \, \partial Y^\mu \bar{\partial} Y^\nu G^{\text{ren}}_{\mu\nu}, \tag{5.116}$$

with

$$G^{\text{ren}}_{\mu\nu} = G^{\text{bare}}_{\mu\nu} - \mu^{-\epsilon} \frac{\alpha'}{3} R_{\mu\nu} \frac{1}{\epsilon}. \tag{5.117}$$

The divergence we have found means that there is a nonvanishing $\beta$ function which is simply the coefficient in front of the divergence $\frac{1}{\epsilon}$:

$$\beta_{\mu\nu} = \mu \frac{\partial}{\partial \mu} G^{\text{ren}}_{\mu\nu} \bigg|_{\epsilon \to 0} \propto \alpha' R_{\mu\nu}, \tag{5.118}$$

namely the $\beta$ function is the spacetime Ricci tensor. Then if we impose the condition

$$\beta_{\mu\nu} = 0, \tag{5.119}$$

we get

$$\boxed{R_{\mu\nu} = 0,} \tag{5.120}$$

which are the vacuum Einstein equations. Therefore we have established that, at one-loop order in the world-sheet QFT, conformal invariance implies the Einstein equations!

If we go on with the Riemann normal coordinates expansion, we get

$$G_{\mu\nu}(X) = G_{\mu\nu}(X_0) - \frac{1}{3} \alpha' R_{\mu\nu\rho\sigma} Y^\rho Y^\sigma + \frac{8}{180} (\alpha')^2 R_{\mu\alpha\beta\lambda} R_{\nu\gamma\delta}^{\ \ \lambda} Y^\alpha Y^\beta Y^\gamma Y^\delta + \mathcal{O}(Y^5). \tag{5.121}$$

At $\alpha'^2$ there is a two-loop self-energy. If we evaluate it we can get the following $\beta$ function:

$$\beta_{\mu\nu} \propto \alpha' R_{\mu\nu} + \sharp (\alpha')^2 R_{\mu\rho} R_\nu^{\ \rho}. \tag{5.122}$$

Then if we set $\beta = 0$ we find the first $\alpha'$ correction to the Einstein equations.

It is instructive to consider what we have just seen from the point of view of Weyl invariance. Using dimensional regularisation we can write

$$S[X] = \int d^{2+\epsilon} z \, \sqrt{\gamma} \gamma^{ab} \partial_a X^\mu \bar{\partial}_b X^\nu G_{\mu\nu}(X), \tag{5.123}$$

where $\gamma_{ab}$ is the worldsheet metric and the indices $a, b = 1, 2$ are the worldsheet indices. In the conformal gauge

$$\gamma^{ab} = e^{+\phi} \delta^{ab}, \tag{5.124a}$$

$$\gamma_{ab} = e^{-\phi} \delta_{ab}, \tag{5.124b}$$

$$\gamma = \det(\gamma_{ab}) = e^{\text{Tr}\{\log \gamma_{ab}\}} = e^{-\phi \, \text{Tr}\{\delta_{ab}\}} = e^{-\phi(2+2\epsilon)}, \tag{5.125}$$

$$\sqrt{\gamma} \gamma^{ab} = e^{-\epsilon \phi} \delta^{ab} = (1 - \epsilon \phi) \delta^{ab}. \tag{5.126}$$

Therefore

$$S[X] = \int d^{2+\epsilon} z \, (1 - \epsilon \phi) \partial X^\mu \bar{\partial} X^\nu G_{\mu\nu}(X). \tag{5.127}$$

If we evaluate the loop contribution and renormalize $G_{\mu\nu}$, we find

$$
\begin{aligned}
S[X] &= \int d^{2+\epsilon} z \, (1 - \epsilon\phi) \partial X^\mu \bar{\partial} X^\nu \left( G_{\mu\nu} - \frac{\alpha'}{3\epsilon} R_{\mu\nu} \right) \\
&\overset{\epsilon \to 0}{=} \int d^2 z \left( \partial X^\mu \bar{\partial} X^\nu G_{\mu\nu} + \alpha' \phi R_{\mu\nu} \partial X^\mu \bar{\partial} X^\nu - \frac{\alpha'}{3\epsilon} R_{\mu\nu} \partial X^\mu \bar{\partial} X^\nu \right).
\end{aligned}
\tag{5.128}
$$

We can then see that the $\phi$ field does not decouple anymore. This results in an anomaly in the Weyl invariance. Indeed

$$
\delta_w S = \frac{\delta S}{\delta \gamma^{ab}} \delta_w \gamma^{ab} = T_{ab}(-w\gamma^{ab}) = -w T_a^a,
\tag{5.129}
$$

which implies

$$
\delta_w S = 0 \quad \Longleftrightarrow \quad T_a^a = 0.
\tag{5.130}
$$

In particular for the action we are considering we have

$$
\begin{aligned}
T^{ab} &= \frac{\delta S}{\delta \gamma^{ab}} = \frac{\delta S}{\delta \left( e^{-\phi} \delta_{ab} \right)} = \frac{\delta S}{\delta \left( (1-\phi)\delta_{ab} \right)} = -\frac{\delta S}{\delta \phi} \delta^{ab} \\
&= -\alpha' R_{\mu\nu} \partial X^\mu \cdot \bar{\partial} X^\nu \delta^{ab},
\end{aligned}
\tag{5.131}
$$

$$
\begin{aligned}
\delta_w S &= -w T_a^a = \alpha' R_{\mu\nu} \partial X^\mu \cdot \bar{\partial} X^\nu w \delta_a^a \\
&\propto w \alpha' R_{\mu\nu} \partial X^\mu \cdot \bar{\partial} X^\nu.
\end{aligned}
\tag{5.132}
$$

This means that

$$
T_a^a = -\alpha' R_{\mu\nu} \partial X^\mu \cdot \bar{\partial} X^\nu,
\tag{5.133}
$$

and hence

$$
T_a^a = 0 \quad \Longleftrightarrow \quad R_{\mu\nu} = 0.
\tag{5.134}
$$

In other words, Weyl symmetry is non-anomalous if and only if the Einstein equations hold.

Everything we discussed in this section came from the fact that we coupled the graviton to the worldsheet. But in the closed string spectrum at level $N = 1$ we found not only the graviton $G_{\mu\nu}$, but also the Kalb-Ramond $B_{\mu\nu}$ and the dilaton $\Phi$.

As we saw, inserting $G_{\mu\nu}$ in the non-linear sigma model we get Einstein's equations; if we then couple $B_{\mu\nu}$ and $\Phi$ as well, we get a generalized sigma model:

$$
S[X] = \frac{1}{2\pi\alpha'} \int d^2\sigma \left[ \sqrt{\gamma} \gamma^{ab} \partial_a X^\mu \bar{\partial}_b X^\nu G_{\mu\nu}(X) + i\epsilon^{ab} \partial_a X^\mu \bar{\partial}_b X^\nu B_{\mu\nu}(X) + \alpha' R^{(2)} \Phi(X) \right].
\tag{5.135}
$$

The first term is the graviton coupling we discussed above. The second term is the Kalb-Ramond coupling: being a differential form it is topological, and hence it does not couple to the WS metric, but to the volume form $\epsilon^{ab}$. Finally, the third term is the dilaton coupling: being a scalar it has no indices and thus couples to $R^{(2)}$, which has no index and is invariant under diffeomorphisms in $d = 2$ just like the other objects inside $S$. The reason why it couples to the 2-dimensional Ricci scalar (and not for example to the identity, which is the 2d cosmological constant) is (classical) Weyl invariance.

Evaluating the $\beta$ functions one finds the classical result

$$\beta^G_{\mu\nu} = \alpha'\left(R_{\mu\nu} + \nabla_\mu\nabla_\nu\Phi - \frac{1}{4}H_{\mu\rho\sigma}H_\nu{}^{\rho\sigma}\right) + \mathcal{O}((\alpha')^2)\,, \tag{5.136a}$$

$$\beta^B_{\mu\nu} = \alpha'\left(-\frac{1}{2}\Phi\nabla_\rho H^\rho{}_{\mu\nu} + \nabla_\rho\Phi H^\rho{}_{\mu\nu}\right) + \mathcal{O}((\alpha')^2)\,, \tag{5.136b}$$

$$\beta^\Phi = \frac{D-26}{6} + \alpha'\left(-\frac{1}{2}\nabla^2\Phi + \nabla_\mu\Phi\nabla^\mu\Phi - \frac{1}{24}H_{\mu\nu\rho}H^{\mu\nu\rho}\right) + \mathcal{O}((\alpha')^2)\,, \tag{5.136c}$$

where $H$ is the field strength associated to the Kalb-Ramond form. The $(D-26)/6$ term in $\beta^\Phi$ is a tadpole which vanishes when $D = 26$.

We thus coupled gravity to a massless 2-form and to a massless scalar. A specific choice of $G_{\mu\nu}$, $B_{\mu\nu}$ and $\Phi$ such that all the $\beta$ functions vanish is called an *exact background* in string theory. An example is the following:

$$\begin{cases} G_{\mu\nu}(X) = \eta_{\mu\nu}\,, \\ B_{\mu\nu}(X) = 0\,, \\ \Phi(X) = \Phi_0 \in \mathbb{R}\,, \\ D = 26\,. \end{cases} \tag{5.137}$$

The corresponding world-sheet action is

$$\begin{aligned} S &= \frac{1}{2\pi\alpha'}\int d^2\sigma\left(\sqrt{\gamma}\gamma^{ab}\partial_a X^\mu\bar{\partial}_b X_\mu + \alpha'\Phi_0 R^{(2)}\right) \\ &= \left(\frac{1}{2\pi\alpha'}\int d^2\sigma\sqrt{\gamma}\gamma^{ab}\partial_a X^\mu\bar{\partial}_b X_\mu\right) + \Phi_0\chi\,, \end{aligned} \tag{5.138}$$

where $\chi$ is the Euler characteristics (for simplicity we are assuming no boundary and no open string deformations).

If we recall that

$$g_s = e^\lambda\,, \tag{5.139}$$

we find that the constant dilaton background implies $\lambda = \Phi_0 = \langle\Phi\rangle$, i.e. the vacuum expectation value of the dilaton:

$$\boxed{g_s = e^{\langle\Phi\rangle}\,.} \tag{5.140}$$

This means there is really no free parameter in string theory. The dilaton is responsible, with its dynamics, of deciding whether string theory is weakly coupled (and thus a perturbative worldsheet description is possible) or strongly coupled, where the worldsheet fades away and non-perturbative effects become important.

A simple example where the dilaton plays a slightly more dynamical role is the so-called *non-critical* string theory. Consider the following background with a linearly rising dilaton

$$\begin{cases} G_{\mu\nu}(X) = \eta_{\mu\nu}\,, \\ B_{\mu\nu}(X) = 0\,, \\ \Phi(X) = \Phi_0 + v_\mu X^\mu\,, \\ D \neq 26\,, \end{cases} \tag{5.141}$$

where $v_\mu$ is a constant vector to be fixed. From the $\beta$ function (5.136c) we see that we need

$$v_\mu v^\mu = \frac{26-D}{6\alpha'}\,. \tag{5.142}$$

It can be shown that the $\beta$ function remains zero to all higher orders in $\alpha'$ and therefore this is another exact string theory background. When $D < 26$ the dilaton gradient $v^\mu$ is a space-like vector. This represents a space-like direction along which the dilaton $\Phi$ grows linearly. Since we have just argued that $g_s \sim e^\Phi = e^{\Phi_0 + v \cdot X}$ the string theory is very strongly coupled for positive large $v \cdot X$ while it is very weakly coupled for negative values. This also teaches us that, at the end of the day, we don't really need $D = 26$ for a consistent string theory, but only $c = 26$. When a linear dilaton background is switched on as in (5.141) it can be shown that the CFT of the scalar $v \cdot X$ has a central charge $c = 26 - D + 1$ and therefore the full matter CFT (including the remaining $D - 1$ free bosons) will still have $c = 26$.

## 5.5 One-loop amplitudes

At the beginning of this chapter we wrote down a master formula to compute a generic $n_c + n_o$ closed-open string amplitude:

$$\mathcal{A}(1, \ldots, n) = \sum_{g,b} g_s^{n_c + n_o/2} \, g_s^{2(g-1)+b} \langle\!\langle \mathcal{V}_1 \cdots \mathcal{V}_{n_c} \, V_1 \cdots V_{n_o} \rangle\!\rangle_{\Sigma_{g,b}}, \tag{5.143}$$

$$\langle\!\langle \mathcal{V}_1 \cdots \mathcal{V}_{n_c} \, V_1 \cdots V_{n_o} \rangle\!\rangle_{\Sigma_{g,b}} = \int \frac{\mathcal{D}h^{(g,b)}\mathcal{D}X}{\text{Vol}(\mathcal{G})} \mathcal{V}_1 \cdots \mathcal{V}_{n_c} \, V_1 \cdots V_{n_o} \, e^{-S_{\text{Pol}}[X, h^{(g,b)}]}. \tag{5.144}$$

If we focus again for a moment on the tree-level amplitude we know that we have to compute a CFT correlator on the sphere or the disk, where the metric $h$ is pure gauge and we can completely fix it to a fiducial value $h_{\alpha\beta} = \eta_{\alpha\beta}$. However when we are at $g > 0$ or $b > 1$ it turns out that the metric has non-trivial degrees of freedom which cannot be changed by Diff$\times$Weyl gauge transformations. These metric degrees of freedom are finite for any given topology $(g, b)$ and are called *Moduli*. The moduli of a Riemann surface are the coordinates of a finite dimensional space $\mathcal{M}_{g,b}$ which is called *Moduli Space*. Calling $t^i$ ($i = 1, \cdots, \dim\mathcal{M}_{g,b}$) the moduli, a fiducial value of the metric will then be a function on the moduli space $\mathcal{M}_{g,b}$

$$\hat{h}_{\alpha\beta} = \hat{h}_{\alpha\beta}(t^i), \tag{5.145}$$

and the situation is now that *any* metric $h_{\alpha\beta}$ at a given value of the moduli can be obtained by doing a Diff$\times$Weyl gauge transformation $\zeta$ on a fiducial value

$$h_{\alpha\beta}(t^i) = \left(\hat{h}_{\alpha\beta}(t^i)\right)^\zeta. \tag{5.146}$$

Given this understanding the functional integral over the metrics still cancels the infinite volume of the gauge group, but a finite dimensional integral over the moduli space is left

$$\int \frac{\mathcal{D}h}{\text{Vol}(\mathcal{G})} = \int \frac{\mathcal{D}^{\text{gauge}}h\,\mathcal{D}^{\text{mod}}h}{\text{Vol}(\mathcal{G})} = \int \frac{\mathcal{D}\zeta}{\text{Vol}(\mathcal{G})} \, d^n t^i \det\left(\frac{\mathcal{D}\hat{h}}{\mathcal{D}\zeta}, \frac{\partial\hat{h}}{\partial t}\right)$$
$$= \int_{\mathcal{M}_{g,b}} d^n t^i \det\left(\frac{\mathcal{D}\hat{h}}{\mathcal{D}\zeta}, \frac{\partial\hat{h}}{\partial t}\right), \tag{5.147}$$

where $n = \dim\mathcal{M}$. The combined functional determinant can be again exponentiated with the FP trick and, after some manipulation which we will not discuss (see any book for details), it gives rise to

$$\det\left(\frac{\mathcal{D}\hat{h}}{\mathcal{D}\zeta}, \frac{\partial\hat{h}}{\partial t}\right) = \int \mathcal{D}b\,\mathcal{D}c \prod_{i=1}^{\dim\mathcal{M}} \left(b, \partial_{t^i}\hat{h}(t)\right) e^{-\frac{1}{4\pi}(b, 2P_1 c)}, \tag{5.148}$$

where we have defined for convenience the scalar product between quadratic symmetric differentials

$$(\psi, \chi) \equiv \int d^2\sigma \sqrt{\hat{h}}\, \psi_{\alpha\beta}\, \hat{h}^{\alpha\gamma} \hat{h}^{\beta\delta}\, \chi_{\gamma\delta}\,, \tag{5.149}$$

and we recall that

$$(P_1 c)_{\alpha\beta} \equiv \frac{1}{2}\left(\hat{\nabla}_\alpha c_\beta + \hat{\nabla}_\beta c_\alpha\right) - \hat{h}_{\alpha\beta}\, \hat{\nabla} \cdot c\,. \tag{5.150}$$

Because of the form of the FP action $(b, P_1 c)$, there are in general ghost *zero modes*. To start with, there can be $c$ zero modes associated to globally defined conformal Killing vectors which are in the Kernel of $P_1$

$$(P_1 c)_{\alpha\beta} = 0 \qquad \text{(Conformal Killing vectors/global } c \text{ zero modes).} \tag{5.151}$$

However there are also global $b$ zero modes. To see this consider

$$(b, P_1 c) = \int d^2\sigma \sqrt{\hat{h}}\, b_{\alpha\beta}(P_1 c)^{\alpha\beta} = \int d^2\sigma \sqrt{\hat{h}}\,(P_1^T b)_\alpha c^\alpha\,, \tag{5.152}$$

where $P_1^T$ sends traceless quadratic differentials to vector fields

$$(P_1^T \delta h)_\alpha = -2\hat{\nabla}^\beta \delta h_{\alpha\beta}\,. \tag{5.153}$$

The *moduli variations* of the metric (i.e. the variations that are not gauge transformations) are precisely defined as the kernel of $P_1^T$, so that they are orthogonal to the gauge variations of the metric, with respect to the scalar product (5.149):

$$P_1^T \delta h = 0 \qquad \text{(Moduli Variations).} \tag{5.154}$$

Then it is clear that to every metric modulus there is an associated $b$-ghost zero mode which is defined as

$$P_1^T b = 0 \qquad \text{(Global } b \text{ zero modes).} \tag{5.155}$$

Just like the $c$ zero modes associated to the CKV, these $b$ zero modes also decouple from the FP action and they should be inserted in the path integral to have a non-vanishing result. Very nicely the FP procedure explicitly includes them via the $(b, \partial_i \hat{h}(t))$ in (5.148).

Therefore in a generic string amplitude there should be as many $b$ insertions as the dimension of moduli space. In addition to this (but only for $(g = 1, b = 0)$ and $(g = 0, b = 0, 1, 2)$) there should be $c$ insertions as well, as many as the number of conformal Killing vectors. For $\chi < 0$ there are no CKV's, so in practice the only new cases where we have to deal with them are the torus and the annulus. CKV's therefore don't play any role in higher loop amplitudes, while moduli are the main characters. Indeed the dimension of the moduli space and the dimension of the CKG are related by the Riemann-Roch Theorem:

$$\dim\mathcal{M}_{g,b} - \dim\text{CKG}_{g,b} = \dim\text{Ker}P_1^T - \dim\text{Ker}P_1 = -3\chi = 6g + 3b - 6\,. \tag{5.156}$$

These general informations are enough for this introductory course. In order to better grasp the concept of moduli and the fundamental role they play in strings amplitudes we will now concretely analyze simple(st) examples at $(g = 1, b = 0)$, the torus and $(g = 0, b = 2)$, the annulus. We will start from the torus.

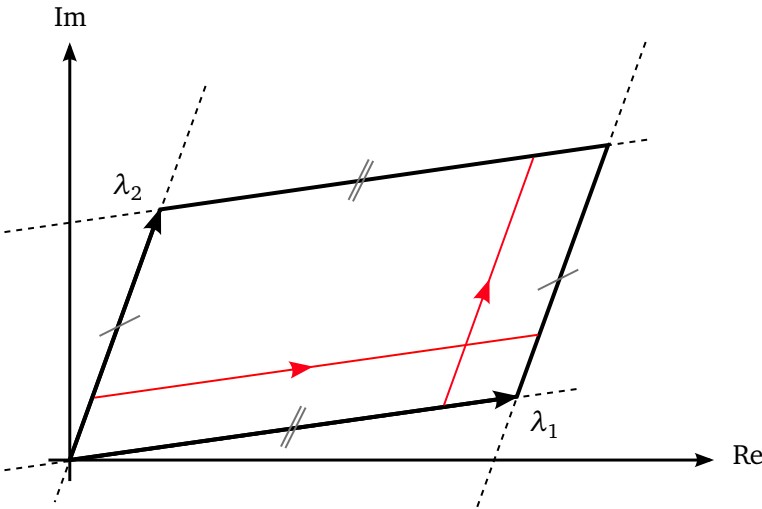

Figure 5.11: Here it is represented one of the cells of the $\Lambda$ lattice. The torus can be obtained by gluing together the opposite sides of the parallelogram. The red lines are an example of nontrivial loops on the surface.

### 5.5.1 Closed string vacuum bubble

A surface of genus one has the topology of a torus. We have then the following theorem.

*Theorem:*
Every closed surface at genus one can be brought to a *flat* torus with a metric $dz d\bar{z}$ with a local Diff $\times$ Weyl transformation, *connected* to the identity.

A flat torus is defined as a particular quotient of $\mathbb{C}$: given two complex numbers $\lambda_1$ and $\lambda_2$, define the identification

$$z \sim z + n\lambda_1 + m\lambda_2, \tag{5.157}$$

with $n, m \in \mathbb{Z}$. This identification can also be expressed using the defining lattice

$$\Lambda = \{\xi \in \mathbb{C} | \xi = n\lambda_1 + m\lambda_2, \ n, m \in \mathbb{Z}\}, \tag{5.158}$$

so that the torus is the quotient $T_2 = \mathbb{C}/\Lambda$.

As we can see in figure 5.11, the torus is characterized by two classes of non-contractible loops (red lines) which are manifestation of the topological non-triviality of this manifold. We say that the torus is flat because the metric is the same as the one on its universal cover, the complex plane, i.e. $dz \otimes_{\text{sym}} d\bar{z}$.

It is useful to introduce a new parameter

$$\tau = \lambda_2/\lambda_1 \in \mathbb{C}, \tag{5.159}$$

called the *modulus* of the torus. Without loss of generality we can assume that $\text{Im}(\tau) > 0$, since if it is not we just rename $\lambda_1 \leftrightarrow \lambda_2$. Notice that $\tau$ is invariant by rescaling the coordinates

$$z \to \alpha z \quad \leftrightarrow \quad \lambda_{1,2} \to \alpha\lambda_{1,2}. \tag{5.160}$$

If we change $z \to \alpha z$, $\alpha \in \mathbb{C}$ we have that the metric scales as $d^2z \to |\alpha|^2 d^2z$ which can be absorbed by a Weyl transformation. From the lattice point of view this rescaling can be seen as a change of the lattice parameters to $\Lambda \to \alpha\Lambda$. This transformation, however, leaves $\tau$ unchanged, so $\tau$ looks like the correct quantity to parameterize the space of inequivalent torii, up to rescalings.

A relevant property of the lattice $\Lambda$ is the existence of an automorphism group $\mathscr{A}$ acting on $\lambda_{1,2}$ that leaves $\Lambda$ invariant

$$\Lambda' = \begin{pmatrix} \lambda_1' \\ \lambda_2' \end{pmatrix} = \begin{pmatrix} a & b \\ c & d \end{pmatrix} \begin{pmatrix} \lambda_1 \\ \lambda_2 \end{pmatrix} = A\Lambda, \tag{5.161}$$

with integer entries $a, b, c, d \in \mathbb{Z}$. This means that the same lattice can be obtained by taking integer linear combinations of $(\lambda_1, \lambda_2)$. However not all linear combinations are allowed, but only those whose inverse is still an integer linear combination

$$A^{-1} = \frac{1}{ad - bc} \begin{pmatrix} d & -b \\ -c & a \end{pmatrix} \in \mathbb{Z}. \tag{5.162}$$

This fixes the value of

$$\det(A) = ad - bc = 1,$$

which defines the group $SL(2, \mathbb{Z})$.

The modulus of the torus is changed by the action of $SL(2, \mathbb{Z})$ with a Möbius transformation

$$\tau' = \frac{\lambda_2'}{\lambda_1'} = \frac{c\lambda_1 + d\lambda_2}{a\lambda_1 + b\lambda_2} = \frac{d\tau + c}{b\tau + a}. \tag{5.163}$$

We now notice that due to the rescaling property (5.160) we can always choose $\alpha = 2\pi/\lambda_1$ so that $(\lambda_1, \lambda_2) \to (2\pi, 2\pi\tau)$. This is a standard parametrization of the torus and, as we can see, is completely fixed by $\tau$.

Can we then say that we get a different torus $\forall \tau \in$ UHP ? The answer is clearly no, because we also have to take into account that two moduli $\tau_1, \tau_2$ related by a $SL(2, \mathbb{Z})$ transformation are describing the same lattice, and thus the same torus. The complete *moduli space* will then be

$$\mathcal{F} = \text{UHP}/SL(2, \mathbb{Z}), \tag{5.164}$$

where $SL(2, \mathbb{Z})$ acts as the group of *modular transformations* (5.163). Let's characterize this portion of the UHP. The $SL(2, \mathbb{Z})$ group is generated by two transformations:

$$\text{Translation } T : \qquad \tau \to \tau + 1 \quad \Longrightarrow T = \begin{pmatrix} 1 & 1 \\ 0 & 1 \end{pmatrix}, \tag{5.165a}$$

$$\text{Inversion } S : \qquad \tau \to -\frac{1}{\tau} \quad \Longrightarrow S = \begin{pmatrix} 0 & -1 \\ 1 & 0 \end{pmatrix}, \tag{5.165b}$$

namely every $A \in SL(2, \mathbb{Z})$ can be expressed as a chain product of the $T$ and $S$ matrices. This means that to quotient the UHP we have to look at the action of the generators:

- the equivalence under $T$ gives $\tau \sim \tau + 1$ so we can just restrict our attention to the band $\text{Re } \tau \in [-1/2, 1/2)$,

- the equivalence under $S$ makes the interior of the unit disk equivalent to its exterior so we can just focus on the values $|z| > 1$.

Under these considerations we see that a fundamental domain (the moduli space of the torus) is as shown in fig. 5.12.

We are now ready to see how different $\tau$'s are associated to different flat metrics on the torus. To do this it is convenient to redefine the complex coordinates of the torus as

$$z = \sigma_1 + \tau\sigma_2, \tag{5.166}$$

$$\bar{z} = \sigma_1 + \bar{\tau}\sigma_2. \tag{5.167}$$

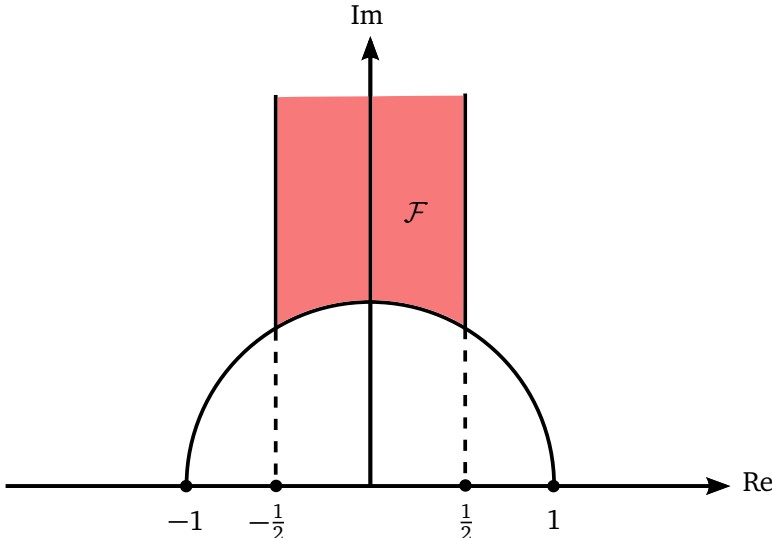

Figure 5.12: The Moduli space $\mathcal{F}$ of the torus. Other equivalent fundamental domains can be obtained by repeated use of the $T$ and $S$ modular transformations on $\mathcal{F}$.

Notice that by construction the coordinates $\sigma_{1,2}$ are *periodic*:

$$\sigma_{1,2} \in (0, 2\pi]. \tag{5.168}$$

As an example, if $\tau = i$ we get the square torus. With this parametrization we can then write the metric on the torus as

$$dz \otimes_{\text{sym}} d\bar{z} = (d\sigma_1 + \tau d\sigma_2) \otimes_{\text{sym}} (d\sigma_1 + \bar{\tau} d\sigma_2), \tag{5.169}$$

notice that the $\tau$ dependence cannot be washed away by a redefinition of the coordinates $\sigma_{1,2}$ because the needed redefinition would spoil their periodicity. So changing $\tau$ results in a genuine different metric. This is thus a very clear example of a family of fiducial metrics in one-to-one correspondence with the two dimensional moduli space with coordinates $(\tau, \bar{\tau})$.

Let us now analyze the effect of modular transformations on the coordinates $(\sigma_1, \sigma_2)$ of the torus. Using the action of $SL(2, \mathbb{Z})$ on $\tau$ we obtain

$$z = \sigma_1 + \tau \sigma_2 \quad \longrightarrow \quad \sigma_1 + \frac{a\tau + b}{c\tau + d} \sigma_2 = \frac{1}{c\tau + d} [(d\sigma_1 + b\sigma_2) + \tau(c\sigma_1 + a\sigma_2)]$$

$$= \frac{1}{c\tau + d} (\sigma'_1 + \tau \sigma'_2). \tag{5.170}$$

If we ignore the effect of the overall factor $c\tau + d$, that is just an unimportant rescaling, we see that $SL(2, \mathbb{Z})$ acts on $(\sigma_1, \sigma_2)$ linearly as a global transformation

$$\begin{pmatrix} \sigma'_1 \\ \sigma'_2 \end{pmatrix} \simeq \begin{pmatrix} d & b \\ c & a \end{pmatrix} \begin{pmatrix} \sigma_1 \\ \sigma_2 \end{pmatrix}. \tag{5.171}$$

Notice that these transformations are not connected to the identity since $a, b, c, d \in \mathbb{Z}$ and that the new coordinates have the same $2\pi$-periodicity as $\sigma_{1,2}$. These discrete transformations are called *global diffeomorphisms* and are part of the gauge group of the worldsheet theory defined on the torus.

After having identified the moduli space, it can be shown that we can finally turn the integral over metrics into an *integral over the moduli space*, i.e.

$$\int \frac{\mathcal{D}h}{Vol(G)} \rightarrow \int_{\mathcal{F}} d^2\tau \, [ghosts]. \tag{5.172}$$

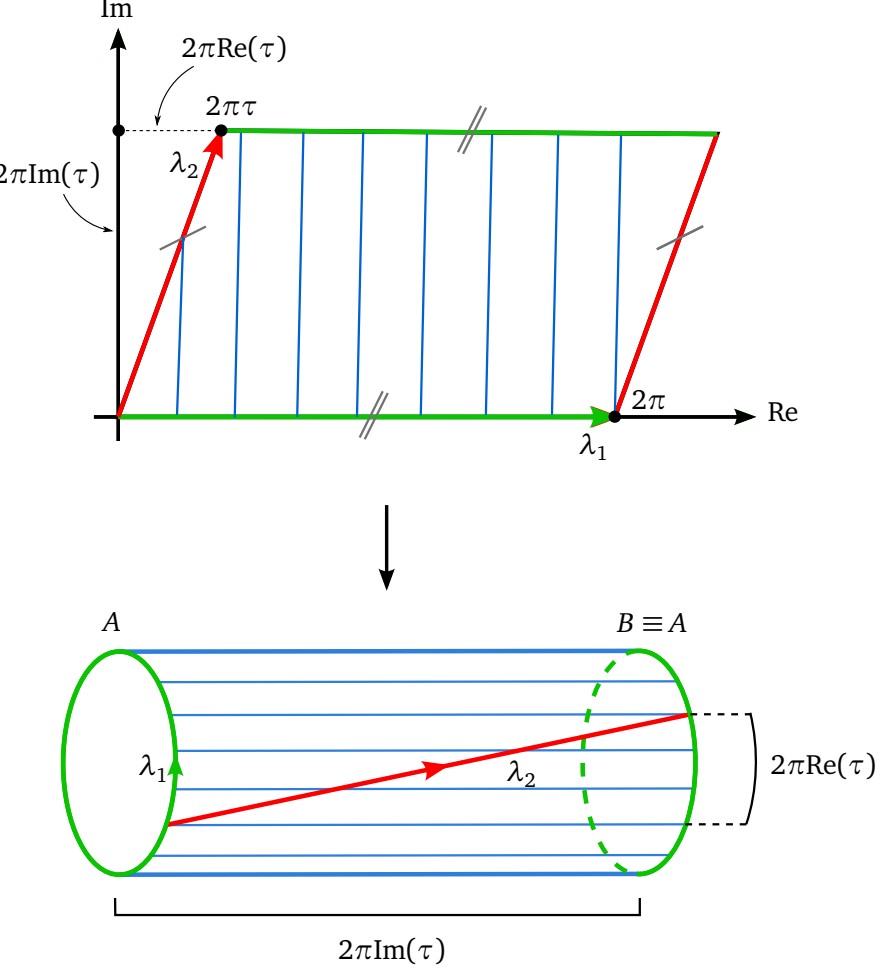

Figure 5.13: Here is depicted the torus in a standard parametrization. Once the parallelogram is glued to form the torus, we can see that $\mathrm{Im}(\tau)$ specifies the length of the torus, namely the ratio of the two main radii $\mathrm{Im}(\tau) = R_1/R_2$ while $\mathrm{Re}(\tau)$ the twist that we have to do on the $B$ cycle before gluing it to the $A$ cycle.

When we reduce the path integral over the metrics into an integral over the moduli space $d\tau d\bar{\tau}$, we must also take into account the volume of the conformal Killing group (CKG) of the torus. This time this is much easier to do than on the sphere, because the volume of the conformal Killing group is finite. There are only two conformal Killing vectors which are the generators of translations along the principal cycles

$$\mathrm{CKG}_{T^2} \sim U(1) \times U(1). \tag{5.173}$$

Because the two circles have finite length, $2\pi$ and $2\pi\mathrm{Im}(\tau)$, we then conclude that the CKG has finite volume equal to $\mathrm{Vol}(\mathrm{CKG}_{T^2}) = 2\pi \times 2\pi\mathrm{Im}(\tau) = 4\pi^2\mathrm{Im}(\tau)$. The overcounting over the CKG will then be taken into account by a normalization factor $\mathrm{Vol}(\mathrm{CKG}_{T^2})^{-1}$ without the need to apply the FP method. In doing this, however, we have to add two zero-mode $c$-insertions to saturate the otherwise vanishing Berezin integral on $c$. An arbitrary $g = 1$ amplitude will then have the following form:

$$\mathcal{A} = \int_{\mathcal{F}} \frac{d^2\tau}{4\pi^2\mathrm{Im}(\tau)} f(\tau, \bar{\tau}), \tag{5.174}$$

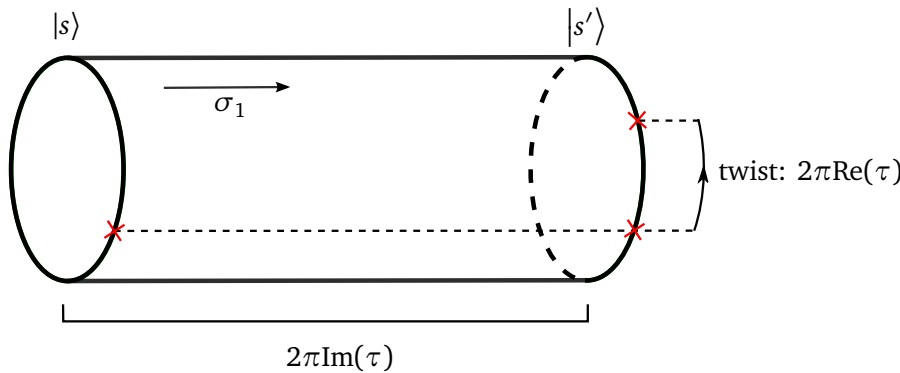

Figure 5.14: State evolution between $|s\rangle$ and $|s'\rangle$.

where

$$f(\tau,\bar{\tau}) = \int \mathcal{D}X\mathcal{D}c\mathcal{D}b\,[B_0\bar{B}_0][C_0\bar{C}_0]\mathcal{V}_1\cdots\mathcal{V}_n\,e^{-S[X,b,c]}. \tag{5.175}$$

Here $[B_0\bar{B}_0]$ are the $b$-ghost insertions associated with the two moduli (see (5.148)) and $[C_0\bar{C}_0]$ are the $c$-ghosts associated with the two conformal Killing vectors. These insertions are needed to have a non vanishing Berezin integral in the zero mode sector which, just as in the $g = 0$ case, decouples from the $b, c$ action.[29]

For simplicity we will focus on the vacuum bubble where we get rid of the complication of all the vertex operators $\mathcal{V}$.

The computation can be further simplified by invoking the already mentioned *lightcone reduction*[30]

$$f(\tau,\bar{\tau}) = \int \mathcal{D}b\mathcal{D}c\mathcal{D}X^+\mathcal{D}X^-\mathcal{D}X^i[B_0\bar{B}_0][C_0\bar{C}_0]e^{-S[X,b,c]} \sim \int \mathcal{D}x_0^{\pm}\mathcal{D}X^i\,e^{-S_{lc}[x_0^{\pm},X^i]}, \tag{5.176}$$

namely the ghosts cancel the oscillator modes of the $X^{\pm}$ coordinates, and we are left with the center of mass coordinates $x_0^{\pm}(t)$ plus the full transverse fields $X^i(t,\sigma)$. This quantity is the partition function on the torus at modulus $\tau$:

$$f(\tau,\bar{\tau}) = \langle[B_0\bar{B}_0][C_0\bar{C}_0]\rangle_{\text{Torus}_\tau}^{\text{(matter+ghosts)}} = \langle 1\rangle_{\text{Torus}_\tau}^{\text{(light cone)}}, \tag{5.177}$$

where we have implemented the light-cone reduction at the level of the CFT correlator that is defined by the path integral. To compute this partition function we will follow an operator method that will make explicit what is happening on the worldsheet.

The basic intuition is that the path integral is computing the partition function of the $c = 24$ CFT of the transverse $X^i$ (together with the zero modes $x_0^{\pm}$) on the torus of modulus $\tau$. If we follow what is represented in fig. 5.14, we can suppose to open the torus along the circle of length $2\pi\text{Im}\tau$. The obtained (twisted) cylinder can be thought of as being obtained from a state $|s\rangle$ with an euclidean time translation $2\pi\text{Im}(\tau)$, generated by the Hamiltonian $H$ and then a rotation around the cylinder axis of an angle $2\pi\text{Re}\tau$, generated by $P$, the worldsheet spatial momentum. The obtained state is then contracted with $\langle s|$ and a complete sum over $s$ is done (this is the operator way of gluing the two circles). In formulas this means

$$f(\tau,\bar{\tau}) = \sum_s \langle s|e^{-2\pi i\text{Re}\tau P}e^{-2\pi\text{Im}(\tau)H}|s\rangle = \text{Tr}\left[e^{-2\pi i\text{Re}\tau P}e^{-2\pi\text{Im}(\tau)H}\right]. \tag{5.178}$$

---

[29]In this case the zero modes of $b$ and $c$ are given by *constant* holomorphic or anti-holomorphic configurations. These are indeed the only globally holomorphic configurations on the torus.

[30]This would not be possible with generic vertex operators inserted.

Now we have to explicitly write down the form of the $P$ and $H$ operators. We know that on the cylinder

$$H = \int \frac{d\sigma}{2\pi} T_{tt}(t,\sigma) = \int \frac{dw}{2\pi i} T(w) + \int \frac{d\bar{w}}{2\pi i} \bar{T}(\bar{w}). \tag{5.179}$$

To express this in terms of known operators we map to the complex plane $z = e^w$, $\bar{z} = e^{\bar{w}}$. In doing this, however, we have to pay attention that the energy-momentum tensor transforms anomalously under conformal transformations

$$T(w) = T(z)z^2 - \frac{c}{24}, \qquad \bar{T}(\bar{w}) = \bar{T}(\bar{z})\bar{z}^2 - \frac{c}{24}. \tag{5.180}$$

The Hamiltonian then becomes a contour integral on the complex plane:

$$\begin{aligned}
H &= \oint \frac{dz}{2\pi i} \frac{1}{z}\left( T(z)z^2 - \frac{c}{24}\right) + \oint \frac{d\bar{z}}{2\pi i} \frac{1}{\bar{z}}\left( \bar{T}(\bar{z})\bar{z}^2 - \frac{c}{24}\right) \\
&= L_0 + \bar{L}_0 - \frac{c}{12}.
\end{aligned} \tag{5.181}$$

Similarly for the momentum we have

$$\begin{aligned}
P &= \int \frac{d\sigma}{2\pi} T_{t\sigma} = \int \frac{dw}{2\pi i} T(w) - \int \frac{d\bar{w}}{2\pi i} \bar{T}(\bar{w}) \\
&= \oint \frac{dz}{2\pi i} \frac{1}{z}\left( T(z)z^2 - \frac{c}{24}\right) - \oint \frac{d\bar{z}}{2\pi i} \frac{1}{\bar{z}}\left( \bar{T}(\bar{z})\bar{z}^2 - \frac{c}{24}\right) \\
&= L_0 - \bar{L}_0.
\end{aligned} \tag{5.182}$$

Plugging these results in the trace we get

$$f(\tau,\bar{\tau}) = \text{Tr}\left[ q^{L_0 - \frac{c}{24}} \bar{q}^{\bar{L}_0 - \frac{c}{24}}\right], \tag{5.183}$$

where we have defined

$$q = e^{2\pi i \tau}. \tag{5.184}$$

Now we can compute the trace using the (light-cone) oscillator basis (3.70):

$$\begin{aligned}
f(\tau,\bar{\tau}) &= |q|^{-\frac{c}{12}} \int \frac{d^{26}P}{(2\pi\sqrt{\alpha'})^{26}} \sum_s \langle s, P| \, |q|^{\frac{\alpha'P^2}{2}} q^N \bar{q}^{\bar{N}} |s, P\rangle \\
&= |q|^{-\frac{c}{12}} \int \frac{d^{26}P}{(2\pi\sqrt{\alpha'})^{26}} |q|^{\frac{\alpha'P^2}{2}} \underbrace{\sum_s \langle s| \, q^N \bar{q}^{\bar{N}} |s\rangle}_{\text{oscillators sum}} \\
&= |q|^{-\frac{c}{12}} \int \frac{d^{26}P}{(2\pi\sqrt{\alpha'})^{26}} |q|^{\frac{\alpha'P^2}{2}} \left[ \prod_{k=1,\bar{k}=1}^{\infty} \sum_{n_k,\bar{n}_{\bar{k}}=0}^{\infty} q^{k n_k} \bar{q}^{\bar{k}\bar{n}_{\bar{k}}}\right]^{24} \\
&= |q|^{-\frac{c}{12}} \int \frac{d^{26}P}{(2\pi\sqrt{\alpha'})^{26}} |q|^{\frac{\alpha'P^2}{2}} \left[ \prod_{k=1}^{\infty} \frac{1}{1-q^k} \prod_{\bar{k}=1}^{\infty} \frac{1}{1-\bar{q}^{\bar{k}}}\right]^{24} \\
&\overset{c=24}{=} \left[ \prod_{k=1}^{\infty} \frac{q^{-\frac{1}{24}}}{1-q^k} \prod_{\bar{k}=1}^{\infty} \frac{\bar{q}^{-\frac{1}{24}}}{1-\bar{q}^{\bar{k}}}\right]^{24} \int \frac{d^{26}P}{(2\pi\sqrt{\alpha'})^{26}} |q|^{\frac{\alpha'P^2}{2}}
\end{aligned}$$

$$= \left[ \prod_{k=1}^{\infty} \frac{q^{-\frac{1}{24}}}{1-q^k} \prod_{\bar{k}=1}^{\infty} \frac{\bar{q}^{-\frac{1}{24}}}{1-\bar{q}^{\bar{k}}} \right]^{24} (\text{Im}(\tau))^{-\frac{26}{2}}$$

$$= \frac{1}{\text{Im}(\tau)} \left[ \frac{1}{\sqrt{\text{Im}(\tau)}} \frac{1}{|\eta(\tau)|^2} \right]^{24}, \tag{5.185}$$

where we have identified the Dedekind eta function $\eta(\tau) = q^{\frac{1}{24}} \prod_{k=1}^{\infty}(1-q^k)$, with $q = e^{2\pi i \tau}$ (see appendix B). In the end we get the following form for the full 1-loop vacuum bubble:

$$\mathcal{A}_{\text{Vacuum}}^{1-\text{loop}} = \int_{\mathcal{F}} \frac{d^2\tau}{(\text{Im}(\tau))^2} \left[ \frac{1}{\sqrt{\text{Im}(\tau)}} \frac{1}{|\eta(\tau)|^2} \right]^{24}. \tag{5.186}$$

We notice that the measure $d\mu = d\tau d\bar{\tau}/(\text{Im}(\tau))^2$ is already modular invariant. Indeed, if we choose

$$\tau' = \frac{a\tau + b}{c\tau + d}, \qquad \bar{\tau}' = \frac{a\bar{\tau} + b}{c\bar{\tau} + d}, \qquad A = \begin{pmatrix} a & b \\ c & d \end{pmatrix} \in SL(2, \mathbb{Z}), \tag{5.187}$$

we have

$$\text{Im}(\tau') = \frac{1}{|c\tau + d|^2} \text{Im}((a\tau + b)(c\bar{\tau} + d))$$

$$= \frac{1}{|c\tau + d|^2} \text{Im}(bc\bar{\tau} + ad\tau)$$

$$= \frac{\det(A)}{|c\tau + d|^2} \text{Im}(\tau) = \frac{\text{Im}(\tau)}{|c\tau + d|^2}, \tag{5.188}$$

$$d\tau' = \frac{d\tau a}{c\tau + d} - \frac{a\tau + b}{(c\tau + d)^2} d\tau c$$

$$= \frac{\cancel{d\tau\tau ac} + d\tau ad - \cancel{d\tau\tau ac} - d\tau bc}{(c\tau + d)^2}$$

$$= \frac{\det(A)}{(c\tau + d)^2} d\tau = \frac{d\tau}{(c\tau + d)^2}, \tag{5.189}$$

so that in the total measure the scaling terms cancel leaving the measure unaltered:

$$\frac{d\tau' d\bar{\tau}'}{(\text{Im}(\tau'))^2} = \frac{|c\tau + d|^4}{(\text{Im}(\tau))^2} \frac{d\tau}{(c\tau + d)^2} \frac{d\bar{\tau}}{(c\bar{\tau} + d)^2} = \frac{d\tau d\bar{\tau}}{(\text{Im}(\tau))^2}. \tag{5.190}$$

This implies that the function:

$$\tilde{f}(\tau, \bar{\tau}) = \left[ \frac{1}{\sqrt{\text{Im}(\tau)}} \frac{1}{|\eta(\tau)|^2} \right]^{24}, \tag{5.191}$$

must be invariant under $SL(2, \mathbb{Z})$. In fact it is not difficult to realize that $\tilde{f}$ is precisely the torus partition function of the $c = 24$ CFT described by the transverse scalars $X^i(z, \bar{z})$, *without* the light-cone zero modes $x^{\pm}(t)$ (whose contribution has been absorbed into the measure to make it modular invariant). *This CFT partition function must be modular invariant.* If this was not the case, it would entail that equivalent torii would give inequivalent contributions with a catastrophic anomaly in the global diffeomorphism invariance.

Modular invariance is a very strong constraint on the spectrum of a CFT, which essentially selects what kind of states can circulate in a string loop. There is no analog of this in a usual QFT and it is ultimately a purely string effect. It is a rather subtle effect, though, that cannot

be seen at tree level. In the case of the bosonic string the function $\tilde{f}$ is fully modular invariant as we will now prove.

First of all consider the following properties of the function $\eta(\tau)$:

$$\eta(\tau+1) = e^{\frac{i\pi}{12}}\eta(\tau), \qquad \eta\left(-\frac{1}{\tau}\right) = \sqrt{-i\tau}\,\eta(\tau). \tag{5.192}$$

While the first one is easy to prove, the second property requires some more work: a proof is provided in appendix B. Now using the modular properties we have

$$\tilde{f}(T(\tau)) = \tilde{f}(\tau+1) = \left[\frac{1}{\sqrt{\text{Im}(\tau)}}\frac{1}{|e^{\frac{i\pi}{12}}\eta(\tau)|^2}\right]^{24} = \tilde{f}(\tau). \tag{5.193}$$

Moreover, using

$$\text{Im}\left(-\frac{1}{\tau}\right) = -\text{Im}\left(\frac{\bar{\tau}}{|\tau|^2}\right) = \frac{\text{Im}(\tau)}{|\tau|^2}, \tag{5.194}$$

we also get

$$\tilde{f}(S(\tau)) = \tilde{f}\left(-\frac{1}{\tau}\right) = \left[\frac{|\tau|}{\sqrt{\text{Im}(\tau)}}\frac{1}{|\sqrt{-i\tau}\eta(\tau)|^2}\right]^{24} = \tilde{f}(\tau). \tag{5.195}$$

Because $\tilde{f}$ is invariant under the generators of the modular group, then it is invariant under the full group. This concludes the proof of the modular invariance of $\tilde{f}$ in the bosonic case.

For the superstring, on the contrary, the modular invariance will not be guaranteed. As we will see, there are four different ways to truncate the superstring spectrum so that the 1-loop partition function preserves the modular invariance. This projection procedure is known as *GSO projection* (Gliozzi, Scherk, Olive) and the above-mentioned four ways define four types of string theories: Type IIA and Type IIB with no tachyon, space-time fermions and space-time supersymmetry, Type 0A and Type 0B with a tachyon and no fermions.

It is often said that the string amplitudes are free of UV divergences. To appreciate it we can draw a parallel between the 1-loop string vacuum bubble and a point particle vacuum bubble.

A worldline loop for a free massive particle in $D$ dimensions can be computed by applying a procedure similar to the operatorial construction done for the closed string vacuum bubble, with the difference that in this case we are dealing with a 1-dimensional object. This means that inequivalent loops are fixed by a single length parameter $t$ (the analog of $\text{Im}\tau$ for the string) so the form of the amplitude boils down to

$$\mathcal{A}_{pp} = \int_0^\infty \frac{dt}{\text{Vol}(CKG)}\sum_s \langle s|e^{-2\pi tH}|s\rangle = \int_0^\infty \frac{dt}{\text{Vol}(CKG)}\text{Tr}\left(e^{-2\pi tH}\right)$$

$$= \int_0^\infty \frac{dt}{2\pi t}\int\frac{d^d p}{(2\pi)^d}e^{-2\pi t(p^2+m^2)} = \int_0^\infty \frac{dt}{2\pi t}\left(\frac{1}{8\pi^2 t}\right)^{\frac{d}{2}}e^{-2\pi tm^2}, \tag{5.196}$$

where we are also taking into account that the volume of the conformal Killing group for the circle is made up just of translations, so $\text{Vol}(CKG) = 2\pi t$ namely the length of the loop. Thus $t$ is basically the 1-dimensional modulus of the loop, analogous to $\text{Im}(\tau)$.

This integral typically diverges in $t \to 0$, which corresponds to the contribution of very high space time momentum circulating in the loop. This is the expected UV divergence of a particle QFT. On the other hand, as long as $m^2 > 0$, the integral converges in $t \to \infty$ which corresponds to very low space-time momenta in the loop, the infrared region. Notice that a vanishing or (even worse) a tachyonic mass would make the integral divergent in the IR. These are IR divergences.

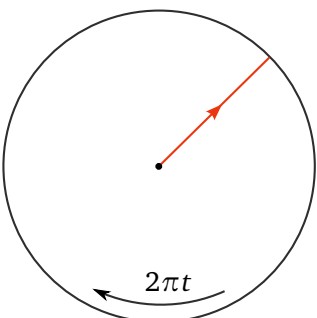

Figure 5.15: A worldline loop for a free massive particle in $d$ dimensions.

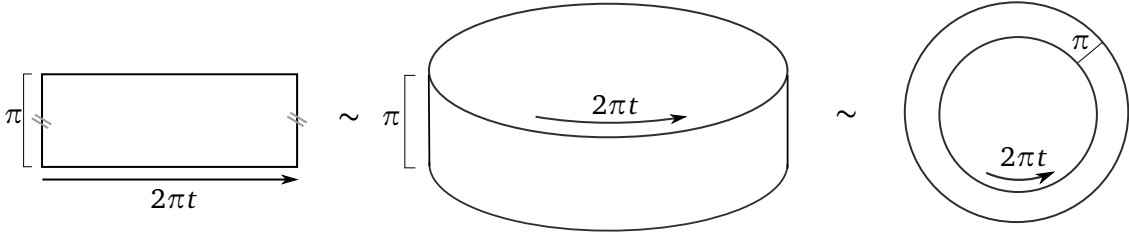

Figure 5.16: The open string vacuum bubble with its single modulus $t$.

Let's compare the UV divergence we get at $t \to 0$ with the closed string case. In this case, inside the integration region $\mathcal{F}$ we have $\mathrm{Im}(\tau) \neq 0$, so that the ultraviolet limit for the closed string loop is always cut-off. This is a key ingredient that makes the string UV finite. In general for $g = 1$ the general structure for an $n$ point amplitude will be

$$\mathcal{A}_{g=1}(1,\ldots,n) \int_{\mathcal{F}} \frac{d^2\tau}{(\mathrm{Im}(\tau))^2} f_n(\tau,\bar{\tau}), \tag{5.197}$$

with $f_n(\tau,\bar{\tau})$ a modular invariant function that depends on the scattering strings. Because the UV region is always excluded in the moduli space this amplitude will always be free of UV divergences.[31]

### 5.5.2 Open string vacuum bubble

Let us now analyze 1-loop open string amplitudes. We consider a 1-loop vacuum bubble that can be described as a cylinder or as an annulus (see fig. 5.16).

It is easy to understand that we have a single modulus $t$ ($\sim$ ratio between the radius and height of the cylinder) analogous to the one of the free particle, and a single CKG generator representing the translations along $t$ or the rotations of the cylinder. The amplitude will then be fully analogous to the particle

$$\mathcal{A}_{\mathrm{Vacuum}}^{1-\mathrm{loop}} = \int_0^\infty \frac{dt}{\mathrm{Vol}(CKG)} \mathrm{Tr}\big[e^{-2\pi tH}\big]^{lc} = \int_0^\infty \frac{dt}{2\pi t} \mathrm{Tr}\big[e^{-2\pi t\left(L_0 - \frac{c}{24}\right)}\big]^{lc}, \tag{5.198}$$

where, as usual, $\mathrm{Tr}[\cdots]^{lc}$ means that we are tracing over transverse oscillators but we do the full $D$-dimensional momentum integral. For the open string inside the trace we just have a

---

[31]Other divergences could however affect the amplitude. For example divergences associated to the collision of vertex operators. These possible divergences correspond to infrared divergences in space-time and need proper treatment as in usual QFT. String field theory provides a systematic approach for these divergences, but this goes beyond the scope of these introductory lectures.

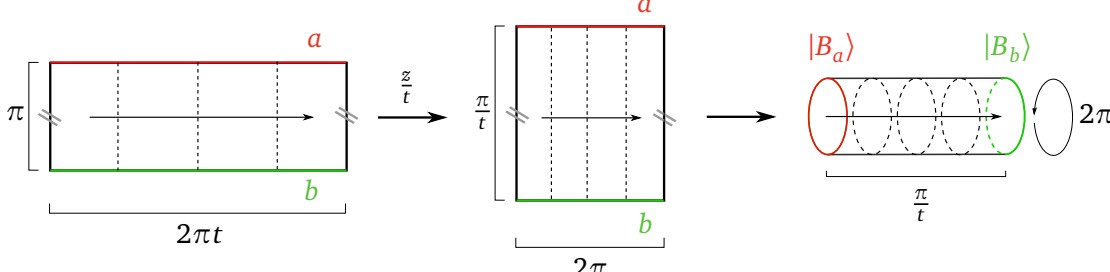

Figure 5.17: The open loop can be interpreted as an untwisted closed string of length $\pi/t$ propagating between two boundary states $|B_a\rangle$ and $|B_b\rangle$.

Hamiltonian term because there is no twisting to be considered. We can notice that for $t \to 0$ the cylinder shrinks to zero radius (UV region), while for $t \to \infty$ the cylinder extends to infinite radius (IR region).

To explicitly compute the trace in (5.198) we fix the background to a single $D(p)$-brane, namely $p+1$ Neumann directions and $D-p-1$ Dirichlet directions (with $D = 26$). In this case we have

$$L_0 = \alpha' P^2 + \hat{N}_N + \hat{N}_D,$$

where $P$ is the momentum on the $p+1$ Neumann directions. Then we have

$$
\begin{aligned}
\mathcal{A}_{\text{Vacuum}}^{1-\text{loop}} &= \int_0^\infty \frac{dt}{2\pi t} \, \text{Tr}\left[ e^{-2\pi t \alpha' p^2} e^{-2\pi t(\hat{N}_N + \hat{N}_D)} e^{2\pi t \frac{D-2}{24}} \right] \\
&= \int_0^\infty \frac{dt}{2\pi t} \int \frac{d^{p+1}P}{(2\pi)^{p+1}} e^{-2\pi t \alpha' P^2} q^{-\frac{D-2}{24}} \left[ \prod_{n=1}^\infty \frac{1}{1-q^n} \right]^{D-2} \\
&\propto \int_0^\infty \frac{dt}{2t} \left( \frac{1}{2t} \right)^{\frac{p+1}{2}} \left[ \frac{1}{\eta(it)} \right]^{D-2}.
\end{aligned}
\tag{5.199}
$$

A similar computation can also be done in the case of 2 parallel $D(p)$-branes separated by a distance $\Delta Y$. In this case we know that $L_0$ gets shifted as follows

$$L_0 = \alpha' P^2 + \left( \frac{\Delta Y}{2\pi\sqrt{\alpha'}} \right)^2 + \hat{N}_N + \hat{N}_D, \tag{5.200}$$

so that the amplitude gets modified by an extra gaussian term:

$$
\begin{aligned}
\mathcal{A}_{\text{Vacuum}}^{1-\text{loop}} &= \int_0^\infty \frac{dt}{(2t)^2} \left( \frac{1}{2t} \right)^{\frac{p-1}{2}} e^{-2\pi t \left[ \frac{\Delta Y}{2\pi\sqrt{\alpha'}} \right]^2} \left[ \frac{1}{\eta(it)} \right]^{D-2} \tag{5.201} \\
&= \int_0^\infty \frac{dt}{(2t)^2} \, \text{Tr}_{\mathcal{H}_{\text{open}}^{ab}} \left[ e^{-2\pi t \left( L_0 - \frac{c}{24} \right)} \right], \tag{5.202}
\end{aligned}
$$

where $\text{Tr}_{\mathcal{H}_{\text{open}}^{ab}}[\cdots]$ is computed in the $c = 24$ BCFT containing only the $D-2$ directions which are transverse to the lightcone. As we have already noticed, the only redundancy we have on the cylinder is given by translations and no other global diffeomorphism, so there is no constraint from modular invariance. This means that in this case the modulus integral runs from 0 to $\infty$ and we are back with an apparent divergent result for $t \to 0$. It seems then that, just as for the particle, we are getting again a UV divergence. However this open string UV divergence can be reinterpreted as a closed string IR divergence due to exchange of closed string tachyons and massless modes. Let's see how this comes about

Figure 5.18: Step-by-step procedure used to compute the boundary state braket.

Indeed we can suppose to consider an open string loop fixing the boundary conditions $F_a(X(0,\tau)) = 0$, $F_b(X(\pi,\tau)) = 0$ and compute

$$\text{Tr}_{\mathcal{H}_{\text{open}}^{ab}}\left[e^{-2\pi t\left(L_0 - \frac{c}{24}\right)}\right] = \int_{\substack{X(\sigma,\tau+2\pi t)=X(\sigma,\tau) \\ F_a(X(0,\tau))=0, F_b(X(\pi,\tau))=0}} \frac{\mathcal{D}X\mathcal{D}h(t)}{\text{Vol}(\mathcal{G})} e^{-S[X,h(t)]}. \tag{5.203}$$

Then if we follow figure (5.17) we can imagine to use the scaling invariance and apply the transformation $z \to z/t$ so that the open loop can be interpreted as a closed string propagating for a worldsheet time $\pi/t$ between two states, implicitly defined by the boundary conditions $a$ and $b$, which we can call $|B_a\rangle$ and $|B_b\rangle$. Such states are called *boundary states* and using them we can write

$$\text{Tr}_{\mathcal{H}_{\text{open}}^{ab}}\left[e^{-2\pi t\left(L_0 - \frac{c}{24}\right)}\right] = \langle B_b| e^{-\frac{\pi}{t}\left(L_0 + \bar{L}_0 - \frac{c}{12}\right)} |B_a\rangle. \tag{5.204}$$

It is obvious then that the UV limit $t \to 0$ becomes an IR limit ($\pi/t \to \infty$), in the closed interpretation where the closed string propagator becomes infinitely long. The boundary states introduced above are not part of the usual closed string spectrum, and can be written as squeezed-like states ($\sim e^{\alpha^\dagger \cdot \Omega \cdot \tilde{\alpha}^\dagger} |0\rangle_{SL(2,\mathbb{C})}$) on the closed string vacuum. We will not be interested in this precise form but we will define them through their BPZ inner product with a generic closed string state $|\mathcal{V}\rangle$

$$\langle B_a|\mathcal{V}\rangle \equiv \langle \mathcal{V}(0,0)\rangle_{\text{Disk}}^{(a)}, \quad \forall \mathcal{V} \in \mathcal{H}_{\text{closed}}, \tag{5.205}$$

where the apex $(a)$ is there to specify that the correlator on the disk must be computed with the $|B_a\rangle$ boundary condition.

For a primary operator of weight $(h, \bar{h})$, $\langle B_a|\mathcal{V}\rangle$ can be explicitly computed using a transformation going from the UHP to the the disk:

$$z \to w = \frac{1 + iz}{1 - iz}, \tag{5.206}$$

$$\bar{z} \to \bar{w} = \frac{1 - i\bar{z}}{1 + i\bar{z}}. \tag{5.207}$$

Then using the transformation property of a primary field we get

$$\mathcal{V}(w, \bar{w}) = \left(\frac{\partial w}{\partial z}\right)^h \left(\frac{\partial \bar{w}}{\partial \bar{z}}\right)^{\bar{h}} \mathcal{V}(z(w), \bar{z}(\bar{w})) \tag{5.208}$$

$$\implies \langle \mathcal{V}(0,0)\rangle_{\text{Disk}}^{(a)} = (2i)^h (-2i)^{\bar{h}} \delta_{h,\bar{h}} \langle \mathcal{V}(i,-i)\rangle_{\text{UHP}}^{(a)}. \tag{5.209}$$

This can be then computed using the doubling trick (see fig. 5.18).

As an example we can compute the overlap with Dirichlet and Neumann boundary states in the case of

$$\mathcal{V}(z, \bar{z}) = j(z)\bar{j}(\bar{z}) = -\frac{2}{\alpha'} \partial X \bar{\partial} X. \tag{5.210}$$

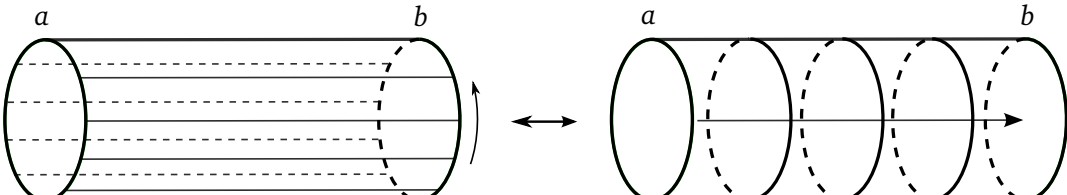

Figure 5.19: Representation of an open string propagating around a loop which, by the open-closed duality, is equivalent to a closed string propagating between two sources $a$ and $b$.

A step by step computation gives

$$
\begin{aligned}
\langle B_{D,N}|\mathcal{V}\rangle &= -\frac{2}{\alpha'}\langle \partial X(0)\bar{\partial}X(0)\rangle^{D,N}_{\text{Disk}} \\
&= -\frac{8}{\alpha'}\langle \partial X(i)\bar{\partial}X(-i)\rangle^{D,N}_{\text{UHP}} \\
&= -\frac{8}{\alpha'}\left(\pm\frac{\alpha'}{2}\frac{1}{(2i)^2}\right) \\
&= \pm 1\,.
\end{aligned}
\tag{5.211}
$$

Going back to (5.204), this a fundamental relation of Boundary Conformal Field Theory, which deserves to be emphasized:

$$
\boxed{\operatorname{Tr}_{\mathcal{H}^{ab}_{\text{open}}}\left[e^{-2\pi t\left(L_0-\frac{c}{24}\right)}\right] = \langle B_b|e^{-\frac{\pi}{t}\left(L_0+\bar{L}_0-\frac{c}{12}\right)}|B_a\rangle\,.}
\tag{5.212}
$$

This states that in any BCFT a 1-loop open string path integral with $a, b$ boundary conditions is equivalent to a closed string propagating between two boundary states $|B_a\rangle, |B_b\rangle$. This is known as the *Cardy condition* and can also be written for the full integrated amplitude as

$$
\boxed{\int_0^\infty \frac{dt}{(2t)^2}\operatorname{Tr}_{\mathcal{H}^{ab}_{\text{open}}}\left[e^{-2\pi t\left(L_0-\frac{c}{24}\right)}\right] = \frac{1}{4\pi}\int_0^\infty ds\,\langle B_b|e^{-s\left(L_0+\bar{L}_0-\frac{c}{12}\right)}|B_a\rangle\,,}
\tag{5.213}
$$

where in the closed string channel we have changed variable to the closed string world-sheet time $s = \frac{\pi}{t}$. This means that a one-loop open string bubble is *the same* as a closed string tree level exchange between two sources, represented by the boundary states.

This is the basic mechanism at the hearth of the *Open-Closed Duality* (OCD) and essentially says that $D$-branes are the sources for closed strings, just like electric charges are the sources for photons. Schematically, a field theory for closed strings in the presence of a $D$-brane represented by boundary conditions $a$ will look like

$$
S \sim \left[\frac{1}{2}\langle\Phi|\,K\,|\Phi\rangle + \langle\Phi|B_a\rangle\right]\,,
\tag{5.214}
$$

where $K$ represents the kinetic operator, while the second term is a source term given by the presence of the $D$-branes, that represent the boundary conditions. This suggests that $D$-branes act as sources for the closed string much in the same way as a current sources the electromagnetic field:

$$
S_{EM} = \int d^d x\left[\frac{1}{4}F_{\mu\nu}F^{\mu\nu} + A_\mu J^\mu\right]\,.
\tag{5.215}
$$

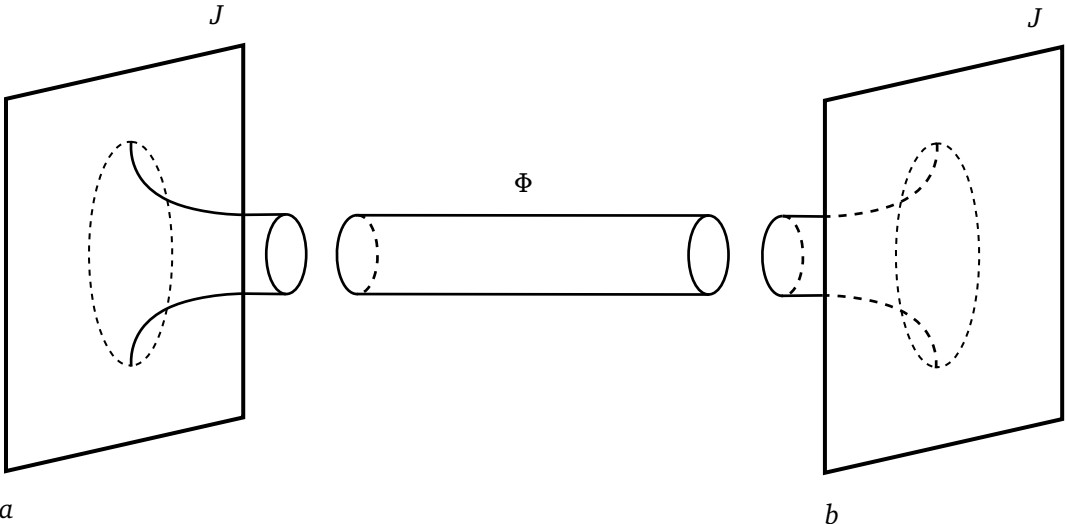

Figure 5.20: A closed string propagating between two $D$-branes which act as sources.

In particular one can focus on the graviton mode of the closed string and realize that the $D$-branes interact with each other by exchanging gravitons.

Making a further conceptual step we can see from (5.214) that a $D$-brane can be understood as a tadpole for the closed string. Therefore the closed string theory will try to cancel the tadpole by going to a new state which in particular means a new space time metric. This is just the *backreaction* of the $D$-brane on the space-time geometry and it is conceptually not different from the Coulomb potential emanating from a charge or the black-hole like metric emanating from a massive source in General Relativity. Proceeding with this line of reasoning we understand then that, in the approximation where closed string theory becomes general relativity (with stringy corrections), the back-reaction of a $D$-brane in space-time should look like a black-hole solution. In the context of the superstring (where the low-energy limit is fully meaningful due to the absence of the tachyon instability) these black-hole like solutions are the so-called *p-branes* of supergravity. Therefore, depending on whether we choose an open or closed string perspective, $D$-branes appear rather differently: they can be thought of as a gauge theory with stringy corrections (open string perspective) or they can be described as a deformed background for the closed strings (closed string perspective). This is the basic idea behind the AdS/CFT correspondence and many other similar phenomena in string theory which go under the name of *open-closed duality*. Unfortunately, because of the omnipresent closed string tachyon, it is not possible to develop this line of reasoning with the critical bosonic string and we need the superstring, or any other open-closed string theory with a stable vacuum.

Let us now analyze explicitly some open string amplitudes, under the assumption that the lightcone momenta along $X^0$, $X^{D-1}$ are integrated out with Neumann conditions and we are left with a $c = 24$ CFT. We thus consider the amplitude

$$
\begin{aligned}
\mathcal{A}_{\text{Vacuum}}^{1-\text{loop}} &= \int_0^\infty \frac{dt}{(2t)^2} \, \text{Tr}_{\text{BCFT}}\left[ e^{-2\pi t(L_0 - 1)} \right] \\
&= \frac{1}{4\pi} \int_0^\infty ds \, \langle B_a | \, e^{-s\left(L_0 + \bar{L}_0 - 2\right)} \, | B_b \rangle \, .
\end{aligned}
\tag{5.216}
$$

We can now analyze the case of a single $D(p)$-brane that was already computed as an integral over $t$:

$$
\begin{aligned}
\mathcal{A}_{\text{Vacuum}}^{1-\text{loop}} &= \int_0^\infty \frac{dt}{(2t)^2} \frac{1}{(2t)^{\frac{p-1}{2}}} \frac{1}{\eta(it)^{24}} = \frac{1}{4\pi} \int_0^\infty ds \left(\frac{s}{2\pi}\right)^{\frac{p-1}{2}} \eta\left(-\frac{\pi}{is}\right)^{-24} \\
&= \frac{1}{4\pi} \int_0^\infty ds \left(\frac{s}{2\pi}\right)^{\frac{p-1}{2}} \left(\frac{\pi}{s}\right)^{-12} \eta\left(\frac{is}{\pi}\right)^{-24} \\
&= \frac{1}{4\pi} \int_0^\infty ds \left(\frac{s}{2\pi}\right)^{\frac{p-25}{2}} \left[\sqrt{2}\,\eta\left(\frac{is}{\pi}\right)\right]^{-24},
\end{aligned}
\tag{5.217}
$$

where we have used the modular property $\sqrt{x}\,\eta(ix) = \eta\left(\frac{i}{x}\right)$. This allows us to identify the $D(p)$-brane boundary states bracket as

$$
\left\langle B_p \right| e^{-s(L_0 + \bar{L}_0 - 2)} \left| B_p \right\rangle = \left(\frac{s}{2\pi}\right)^{\frac{p-25}{2}} \left[\sqrt{2}\,\eta\left(\frac{is}{\pi}\right)\right]^{-24}.
\tag{5.218}
$$

We notice that in the $t$ coordinate the convergence of the amplitude is due to a factor related to the world-volume of the $D$-brane, i.e. $(\sqrt{2t})^{p+1}$. On the contrary for the $s$ coordinate, the closed string variable, the convergence is controlled by the volume of the transverse coordinate, i.e. $(\sqrt{2s})^{p-25}$.

The same computation can be also repeated for a system of 2 $D(p)$-branes separated by a distance $\Delta Y$. The result is the same as for the single $D(p)$-brane case plus the already discussed Gaussian factor:

$$
\left\langle B_p \right| e^{-s(L_0 + \bar{L}_0 - 2)} \left| B_p^{\Delta Y} \right\rangle = \left(\frac{s}{2\pi}\right)^{\frac{p-25}{2}} e^{-\frac{\Delta Y^2}{2s\alpha'}} \left[\sqrt{2}\,\eta\left(\frac{is}{\pi}\right)\right]^{-24}.
\tag{5.219}
$$

To better understand where this term comes from, we can write the open string one-loop bubble in the case of a system of multiple $D(p)$-branes, described by a Chan-Paton factor

$$
\begin{pmatrix} \phi_{aa} & \phi_{ab} \\ \phi_{ba} & \phi_{bb} \end{pmatrix}.
\tag{5.220}
$$

Thus we have

$$
\begin{aligned}
\mathcal{A}_{\text{Vacuum}}^{1-\text{loop}} &= \mathbb{Tr}_{CP}\left[e^{-2\pi t\left(L_0 - \frac{c}{24}\right)}\right] = \mathbb{Tr}_{CP}\left[\begin{pmatrix} \langle aa| & \langle ab| \\ \langle ba| & \langle bb| \end{pmatrix} q^{L_0 - \frac{c}{24}} \begin{pmatrix} |aa\rangle & |ab\rangle \\ |ba\rangle & |bb\rangle \end{pmatrix}\right] \\
&= \text{Tr}_{aa}\left[q^{L_0 - \frac{c}{24}}\right] + \text{Tr}_{bb}\left[q^{L_0 - \frac{c}{24}}\right] + 2\,\text{Tr}_{ab}\left[q^{L_0 - \frac{c}{24}}\right] \\
&= \left\langle B_a \right| e^{-s\left(L_0 + \bar{L}_0 - \frac{c}{12}\right)} \left| B_a \right\rangle + \left\langle B_b \right| e^{-s\left(L_0 + \bar{L}_0 - \frac{c}{12}\right)} \left| B_b \right\rangle + 2\left\langle B_a \right| e^{-s\left(L_0 + \bar{L}_0 - \frac{c}{12}\right)} \left| B_b \right\rangle.
\end{aligned}
\tag{5.221}
$$

We can see that the amplitude splits between self-interaction terms (i.e. $(aa)$, $(bb)$) and interactions terms between different $D$-branes (i.e. $(ab)$, $(ba)$).

### 5.5.3  Forces between $D$-branes

It is important to understand the type of divergences we have in the game. To this end we can consider again the open vacuum bubble between two separated $D(p)$-branes in the $s$ parametrization:

$$
\mathcal{A} = \frac{1}{4\pi} \int_0^\infty ds \left(\frac{2\pi}{s}\right)^{\frac{d_\perp}{2}} e^{-\frac{\Delta Y^2}{2\alpha' s}} \left(\sqrt{2}\,\eta\left(\frac{is}{\pi}\right)\right)^{-24}.
\tag{5.222}
$$

SciPost Phys. Lect. Notes 90 (2025)

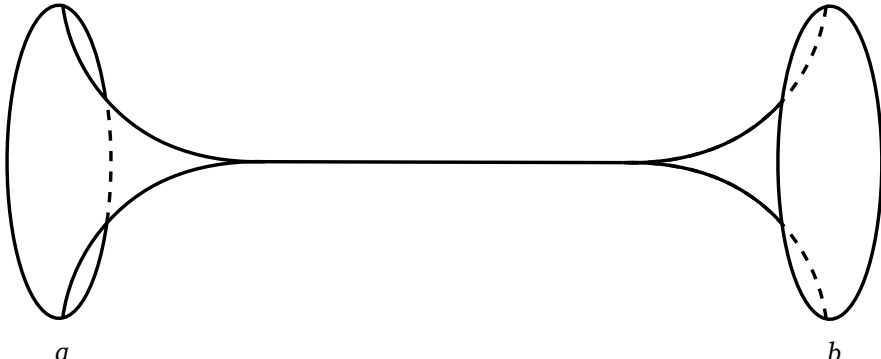

$$a \qquad\qquad\qquad\qquad\qquad\qquad\qquad\qquad\qquad\qquad\qquad b$$

Figure 5.21: Closed string degeneration in the $s \to \infty$ limit.

We see here that the $D$-brane separation acts as a natural cutoff of the $s \to 0$ limit, where the closed string propagator becomes very short and is thus interpreted as the UV of the closed string (which in turns corresponds to the IR of the open string because $s \sim 1/t$).

Focusing on the $s \to \infty$ limit (see fig. 5.21) we can expand the Dedekind eta as follows:

$$\eta\left(\frac{is}{\pi}\right) \sim e^{2s} + 24 + 324\,e^{-2s} + \dots \tag{5.223}$$

As already seen in the lightcone quantization, this is basically an expansion of the form $(\#d.o.f.\ at\ level\ N)\,e^{2(N-1)s}$. We recognise the exponential divergence introduced by the tachyon at level $N = 0$. The next 24 massless d.o.f. come from the propagation of the graviton and the dilaton (due to the requirement that the closed string state should have a non-vanishing disk one-point function, only the diagonal part of the massless vertex operator contributes. In particular the Kalb-Ramond field is not exchanged). Neglecting the tachyonic mode, that will be absent in superstring, the first relevant order is given by the massless string states where $\eta = \text{cost} = 24$, so the amplitude takes the simplified form

$$\mathcal{A} \sim \frac{1}{4\pi} \int_0^\infty ds \left(\frac{2\pi}{s}\right)^{\frac{d_\perp}{2}} e^{-\frac{\Delta Y^2}{2\alpha' s}} (24) = G(\Delta Y), \tag{5.224}$$

where we now specify the dependence over $\Delta Y$ because we want to underline the interpretation of the two $D(p)$-branes as closed strings sources, namely the fact that $G(\Delta Y)$ is computing the source correlator $\langle J(0)J(\Delta Y)\rangle$. To evaluate the integral we substitute $t = 1/s$ so that

$$\begin{aligned} G(\Delta Y) &= \frac{6}{\pi} \int_0^\infty \frac{dt}{t^2} (2\pi t)^{\frac{d_\perp}{2}} e^{-t\frac{\Delta Y^2}{2\alpha'}} \\ &= 12(2\pi)^{\frac{d_\perp}{2}-1} \int_0^\infty dt\, t^{\frac{d_\perp}{2}-2} e^{-t\frac{\Delta Y^2}{2\alpha'}}. \end{aligned} \tag{5.225}$$

Then we implement the change of variable $x = t\frac{\Delta Y^2}{2\alpha'}$ and this gives

$$\begin{aligned} G(\Delta Y) &= 12(2\pi)^{\frac{d_\perp}{2}-1} \left(\frac{2\alpha'}{\Delta Y^2}\right)^{\frac{d_\perp}{2}-1} \int_0^\infty dt\, x^{\frac{d_\perp}{2}-2} e^{-x} \\ &= 12\left(\frac{4\pi\alpha'}{\Delta Y^2}\right)^{\frac{d_\perp}{2}-1} \Gamma\left(\frac{d_\perp}{2}-1\right) \propto \frac{1}{\Delta Y^{d_\perp-2}}. \end{aligned} \tag{5.226}$$

Notice that what we have obtained is essentially the Coulomb/Newton potential between two charges/masses.

This is indeed the same result that we get in electromagnetism coupled to a current

$$S = \int d^d x \left[ \frac{1}{4} F_{\mu\nu} F^{\mu\nu} - j_\mu A^\mu \right], \tag{5.227}$$

in the presence of pointlike charges $q_1$ and $q_2$ at spatial positions $\vec{y}_1$ and $\vec{y}_2$, which means

$$j_\mu(x) = j_\mu^{(1)}(x) + j_\mu^{(2)}(x), \tag{5.228}$$

$$j_\mu^{(i)}(x) = q_i \delta_{\mu,0}\, \delta^{d_\perp}(\vec{x} - \vec{y}_i) = q_i \delta_{\mu,0} \int d^{d_\perp}\vec{k}\, e^{i\vec{k}\cdot(\vec{x} - \vec{y}_i)}, \qquad i = 1,2. \tag{5.229}$$

We can compute the current-current correlation function by standard Feynman rules using the photon propagator in Lorentz gauge $\sim \delta(k_1 + k_2)\frac{\eta^{\mu\nu}}{k_1^2}$

$$G(\Delta x) = \left\langle j^{(1)}(\vec{y}_1) j^{(2)}(\vec{y}_2) \right\rangle \sim q_1 q_2 \int d^{d_\perp}\vec{k}\, e^{i\vec{k}\cdot\vec{\Delta y}} \frac{1}{\vec{k}^2}, \tag{5.230}$$

where we have used the fact that the external momenta have only spatial components and we have absorbed away the infinite volume factor from the time component of the $\delta(k_1 + k_2)$ in the photon propagator. To compute the momentum integral we first give a Schwinger representation of the propagator

$$\frac{1}{\vec{k}^2} = \int_0^\infty ds\, e^{-s\vec{k}^2}, \tag{5.231}$$

and we then perform the gaussian integral in the momenta by completing the square, which gives

$$G(\Delta y) \sim q_1 q_2 \int_0^\infty ds\, \frac{1}{s^{\frac{d_\perp}{2}}} e^{-\frac{\Delta y^2}{4s}}. \tag{5.232}$$

This is fully analogous to (5.224), so we get in the same way

$$G(\Delta y) \sim \frac{q_1 q_2}{|\Delta y|^{d_\perp - 2}}, \tag{5.233}$$

which is the Coulomb potential between two point-like charges.

We can get a similar result in linearized gravity:

$$S = \int d^d x \left[ -\frac{1}{2} h_{\mu\nu} \nabla_{\mu\nu\rho\lambda} h^{\rho\lambda} - T_{\mu\nu} h^{\mu\nu} \right], \tag{5.234}$$

where

$$\nabla^{\mu\nu}_{\ \ \rho\lambda} = \left( \delta^{(\mu}_\rho \delta^{\nu)}_\lambda - \eta^{\mu\nu}\eta_{\rho\lambda} \right) \Box - 2\delta^{(\mu}_{(\rho} \partial^{\nu)}\partial_{\lambda)} + \eta_{\rho\lambda}\partial^\mu\partial^\nu + \eta^{\mu\nu}\partial_\rho\partial_\lambda, \tag{5.235}$$

by fixing the total stress energy tensor to

$$T^{\mu\nu}(x) = m_1\, \eta^{\mu 0}\eta^{\nu 0}\delta(\vec{x} - \vec{y}_1) + m_2\, \eta^{\mu 0}\eta^{\nu 0}\delta(\vec{x} - \vec{y}_2). \tag{5.236}$$

Then using the graviton propagator in the harmonic gauge we get the Newton potential

$$V(\Delta y) = m_1 m_2 \int d^{d_\perp}\vec{p}\, \frac{e^{i\vec{p}\cdot\vec{\Delta y}}}{|\vec{p}|^2} \propto \frac{m_1 m_2}{|\Delta y|^{d_\perp - 2}}. \tag{5.237}$$

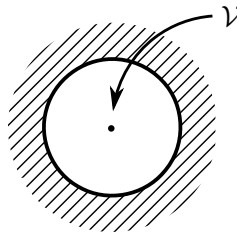

(a) *D-brane interaction, disk insertion.*

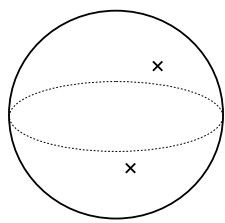

(b) *Kinetic term, 2-vertex insertion on the sphere.*

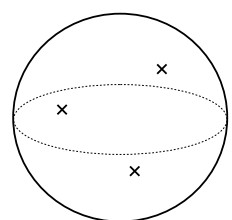

(c) *Typical interaction term, 3-vertex insertion on the sphere.*

Figure 5.22: Relevant correlators for the graviton effective action at tree level.

Notice that these simple QFT calculations are equally valid if the charges or masses are uniformly diffused on $p$-dimensional hyperplanes with $d_\perp$ transverse directions. In this case, of course, the charges $q_i$ will be substituted by charge densities $\mu_i \equiv q_i/\text{Vol}_p$ and the masses $m_i$ by *tensions* $\tau_i \equiv m_i/\text{Vol}_p$. This is precisely the case of $Dp$ branes we have analyzed.

Coming back to the dilaton-gravity interaction between $D(P)$-branes (see eq. 5.226) it is clear that we should be able to interpret the constant factor in front of $1/\Delta Y^{d_\perp - 2}$ as containing the product of the two $D(P)$-brane tensions. This analysis requires a careful comparison with a dilaton-graviton exchange between two sources and it is done in detail in Polchinski 8.7. We will instead give a more qualitative understanding on the D-brane tension mainly concentrating on its dependence on the string coupling constant. It is clear that from the above analysis the $D$-brane tension should be proportional to the amplitude of a graviton emission from a disk as this precisely corresponds to the $h_{\mu\nu}T^{\mu\nu}$ coupling in the effective action. We can then argue that being $\tau_p$ proportional to this 1-point function we will have $\tau_p \sim g_s^{2(g-1)+b+n_c} = g_s^{2-1-1} = g_s^0$ namely the $D$-brane tension seems to not depend on the string coupling constant. However there is a subtlety in this reasoning related to the assignment of the coupling constants in the Polyakov path integral and the assignment of coupling constants in the effective theory. Let's analyze this by constructing the tree level graviton effective action. Because the graviton is a propagating degree of freedom the action will contain a kinetic term of the form $h\hat{K}h$ related to the insertion of two graviton vertex operators on the sphere. This first term will then be proportional to $g_s^{2-2} = 1$ namely it will be independent of the string coupling. Then we can add interactions starting from a cubic $h^3$ term and going on with $h^n$ terms corresponding to the insertions of $n$ vertex operators on the sphere. In general all these terms will have a string coupling dependence of the form $g_s^{n-2}$. If we also include the coupling of a closed string to the disk that we have just discussed, the obtained action is of the form

$$\mathcal{L}_{eff}(h) = \tilde{T}_{\mu\nu}h^{\mu\nu} + hKh + g_s h^3 + g_s^2 h^4 + \dots \tag{5.238}$$

However, if we want to identify the real $T_{\mu\nu}$, we have to match this action with the Einstein-Hilbert action

$$\mathcal{L}_{EH} \sim \frac{R}{g_s^2} + T_{\mu\nu}\hat{h}^{\mu\nu} = \frac{1}{g_s^2}(\hat{h}K\hat{h} + \hat{h}^3 + \hat{h}^4 + \dots) + T_{\mu\nu}\hat{h}^{\mu\nu}, \tag{5.239}$$

with $R = h\hat{K}h + h^3 + h^4 + \dots$ and with Newton constant (which appears in front of the Einstein-Hilbert action) identified (modulo dimensionful constants) as $G_N \sim g_s^2$. This means that the field $h$ in the effective action (5.238) is not normalized in the same way as (5.239) and we must redefine $h \to \hat{h}/g_s$ so that the actual effective action becomes

$$\mathcal{L}_{eff}(\hat{h}) = \frac{1}{g_s}\tilde{T}_{\mu\nu}\hat{h}^{\mu\nu} + \frac{1}{g_s^2}\left[\hat{h}K\hat{h} + \hat{h}^3 + \hat{h}^4 + \dots\right]. \tag{5.240}$$

The actual matter coupling will then be proportional to a $1/g_s$ term which implies

$$\tau_p \propto \frac{1}{g_s}.$$

This means that $D$-branes are *solitons*, i.e. non-perturbative objects which are very heavy at weak coupling and light at strong coupling.

However, given a field theory with coupling constant $g$, solitons typically have an energy that goes like $\sim 1/g^2$. This is easily understandable by the fact that the fields can always be rescaled in such a way that an overall factor of $1/g^2$ appears in front of the action and the fact that solitons are nothing but classical solutions to the field equations, having thus an action (and therefore an energy) which goes as $1/g^2$. For $D$-branes we find instead $1/g_s$.

Another puzzle is that in (5.240) we don't see the $D$-brane as a classical solution, but as a source term $T_{\mu\nu}$. These two puzzles are in fact related and solved in the same way. In string theory we have two basically independent propagating degrees of freedom: open and closed strings. While the closed string coupling constant is naturally $g_{\text{closed}} = g_s$ the open string coupling constant is instead $g_{\text{open}} = g_s^{\frac{1}{2}}$. A full complete space-time theory should be an open-closed theory which structurally would look like

$$
\begin{aligned}
S_{oc}(\Psi_o, \Psi_c) = &\frac{1}{g_s}\left(\frac{1}{2}\langle\Psi_o|\,K_o\,|\Psi_o\rangle + (\text{open string interactions})\right)\\
&+ \frac{1}{g_s^2}\left(\frac{1}{2}\langle\Psi_c|\,K_c\,|\Psi_c\rangle + (\text{closed string interactions})\right)\\
&+ \frac{1}{g_s}\Big(\langle B|\Psi_c\rangle + (\text{open-closed string interactions})\Big),
\end{aligned}
\tag{5.241}
$$

where $\langle B|$ is the boundary state corresponding to the $D$-brane system where the open strings $\Psi_o$ live. Notice that the powers of the string coupling constant are consistent with $g_{\text{open}} = g_s^{\frac{1}{2}}$ and $g_{\text{closed}} = g_s$. One should think of this action as a sort of Yang-Mills theory (open strings) coupled to a sort of gravity (closed strings) where the last line contains the gauge-gravity couplings, in particular the $\langle B|\Psi_c\rangle$ which is analogous of the YM energy momentum tensor coupled to the graviton. From this action it is clear what kind of classical solutions would have an action $\sim 1/g_s$: they are the solutions of the purely open string theory which is what one gets by simply setting to zero the closed strings $\Psi_c$. Suppose we have such a solution $\Psi_o^*$, then obviously its action will depend on $g_s$ like the tension of a $D$-brane

$$S_{oc}(\Psi_o^*, \Psi_c \equiv 0) \sim \frac{1}{g_s}.\tag{5.242}$$

What is the physical interpretation of this solution? It is a new vacuum for the open string, therefore it must be a *new D-brane*. Expanding the open string field around this solution will have the effect (thanks to open-closed interactions which couple multiple open strings to a single closed string) of shifting the original boundary state to a new one

$$\Psi_o \to \Psi_o^* + \varphi_o, \qquad \longrightarrow \qquad |B\rangle \to |B_*\rangle = |B\rangle + \big|\text{ interactions with } \Psi_o^*\big\rangle.\tag{5.243}$$

Therefore it is natural to interpret $D$-branes as *open string solitons*. This seems to be at odds with the common understanding that $D$-branes can be seen in (super)gravity as classical vacuum solutions, without the need of coupling open string matter, just like black-holes in General Relativity can be found without coupling GR to matter. This is in fact a bit misleading: indeed while it is true that the metric of a black-hole solves vacuum Einstein equations everywhere except at the singularity, it is precisely at the singularity (where gravity itself breaks down)

that the source is hidden. If we had a more complete theory we would indeed see that the singularity is in fact a source of the gravitational field, just like a point-like charge is a source of the electromagnetic field (and indeed the electric field diverges at the locus of the charge, just like the curvature of the BH solution does). For the electric charge we know that this divergence is resolved by quantum effects, for gravity we simply don't know how to account for these quantum effects.

This line of reasoning naturally brings to another question. If $D$-branes are open string solitons with a mass $\sim 1/g_s = 1/g_{\text{open}}^2$, why don't we also have *closed string solitons* with a mass $\sim 1/g_s^2 = 1/g_{\text{closed}}^2$? Indeed we do! The most famous example (in the superstring) is the so-called $NS(5)$-brane. This is a $(5+1)$-dimensional extended object. It could be visualized as a sort of $D(5)$-brane, but radically different from it, there are not the usual open strings on it incarnating the physical fluctuations of the soliton. These are still very mysterious objects of which we don't know so much.

# 6 Superstring

## 6.1 Superconformal field theory

The bosonic string we analyzed so far has two problems: it contains no space-time fermions and it has no stable vacuum (since we always find a tachyon in the spectrum). The superstring solves both problems at once, as we will see.[32]

### 6.1.1 Matter $(X; \psi)$-SCFT

In the bosonic matter theory we had the worldsheet bosons $X^\mu(z, \bar{z})$ with $\mu = 0, 1, \ldots, d-1$. Let us now add the primary fields $\psi^\mu(z)$, $\bar{\psi}^\mu(\bar{z})$ of weight $(\frac{1}{2}, 0)$ and $(0, \frac{1}{2})$ respectively. As shown by their conformal weight, they are fields of spin $\frac{1}{2}$ and by spin-statistic we take them to be Grassmann odd.[33]

The corresponding worldsheet action reads

$$S = \frac{1}{4\pi} \int_{\text{WS}} d^2 z \left( \frac{2}{\alpha'} \partial X^\mu \bar{\partial} X_\mu + \psi^\mu \bar{\partial} \psi_\mu + \bar{\psi}^\mu \partial \bar{\psi}_\mu \right), \tag{6.1}$$

which is a free two-dimensional CFT. Such a theory is defined by the following OPEs:

$$X^\mu(z, \bar{z}) X^\nu(0, 0) = -\frac{\alpha'}{2} \eta^{\mu\nu} \log |z| + \text{(regular terms)}, \tag{6.2a}$$

$$\psi^\mu(z) \psi^\nu(0) = \eta^{\mu\nu} \frac{1}{z} + \text{(regular terms)}, \tag{6.2b}$$

$$\bar{\psi}^\mu(\bar{z}) \bar{\psi}^\nu(0) = \eta^{\mu\nu} \frac{1}{\bar{z}} + \text{(regular terms)}. \tag{6.2c}$$

The matter stress-energy tensor is

$$T^{(\text{m})}(z) = -\frac{1}{\alpha'} :\partial X \cdot \partial X: -\frac{1}{2} :\psi \cdot \partial \psi:, \tag{6.3a}$$

$$\overline{T}^{(\text{m})}(\bar{z}) = -\frac{1}{\alpha'} :\bar{\partial} X \cdot \bar{\partial} X: -\frac{1}{2} :\bar{\psi} \cdot \bar{\partial} \bar{\psi}:. \tag{6.3b}$$

One can then show (recall eq. (4.66)) that

$$T^{(\text{m})}(z_1) T^{(\text{m})}(z_2) = \frac{c}{2} \frac{1}{(z_1 - z_2)^4} + \frac{2 T^{(\text{m})}(z_2)}{(z_1 - z_2)^2} + \frac{\partial T^{(\text{m})}(z_2)}{(z_1 - z_2)} + \text{(regular terms)}, \tag{6.4}$$

and the same holds for the anti-holomorphic part. The central charge of the theory is

$$c = c_X + c_\psi = d + \frac{1}{2} d = \frac{3}{2} d, \tag{6.5}$$

where we have added up the contributionsfrom the $X$-CFT and from the $\psi$-CFT.

---

[32]It is not unconceivable that a stable vacuum of bosonic string theory may exist as a classical solution of closed string field theory. Then the bosonic string could be reconsidered.

[33]Given a conformal field of weights $(h, \bar{h})$, its *spin* is $s = h - \bar{h}$.

**Exercise 6.1.1**

Prove eq. (6.4).

Coming back to the worldsheet action (6.1), we have that it is invariant under two possible transformations.

- Conformal transformations ($\infty$-dimensional), which at the infinitesimal level read as

$$
\begin{cases}
\delta_\epsilon^c X^\mu(z,\bar z) &= -\big[\epsilon(z)\partial X^\mu(z,\bar z) + \bar\epsilon(\bar z)\bar\partial X^\mu(z,\bar z)\big]\,, \\
\delta_\epsilon^c \psi^\mu(z) &= -\big[\epsilon\partial\psi^\mu(z) + \tfrac{1}{2}(\partial\epsilon)\psi^\mu(z)\big]\,, \\
\delta_\epsilon^c \bar\psi^\mu(\bar z) &= -\big[\bar\epsilon\bar\partial\bar\psi^\mu(\bar z) + \tfrac{1}{2}(\bar\partial\bar\epsilon)\bar\psi^\mu(\bar z)\big]\,,
\end{cases}
\tag{6.6}
$$

where the superscript "c" stands for "conformal" and where $\epsilon(z)$ is a vectorial field that can be expanded as

$$
\epsilon(z) = \sum_n \epsilon_n z^{-n+1}\,,
\tag{6.7}
$$

namely as a weight $-1$ field.

- Superconformal transformations ($\infty$-dimensional), which at the infinitesimal level read as

$$
\begin{cases}
\delta_\eta^{sc} X^\mu(z,\bar z) &= \sqrt{\tfrac{\alpha'}{2}}\big[\eta(z)\psi^\mu(z) + \bar\eta(\bar z)\bar\psi^\mu(\bar z)\big]\,, \\
\delta_\eta^{sc} \psi^\mu(z) &= \sqrt{\tfrac{2}{\alpha'}}\big[-\eta(z)\partial X^\mu(z)\big]\,, \\
\delta_\eta^{sc} \bar\psi^\mu(\bar z) &= \sqrt{\tfrac{2}{\alpha'}}\big[-\bar\eta(\bar z)\bar\partial\bar X^\mu(\bar z)\big]\,,
\end{cases}
\tag{6.8}
$$

where the superscript "sc" stands for "superconformal". We immediately see that these superconformal transformations create a bridge between the $X$ and the $\psi$ sectors of the CFT, i.e. between bosonic and fermionic fields. This is possible because of the fermionic nature of $\eta(z)$, which can be expanded as

$$
\eta(z) = \sum_r \eta_r z^{-r+\frac{1}{2}}\,,
\tag{6.9}
$$

namely as a weight $-\frac{1}{2}$ field.

Let us now check that the action (6.1) is invariant under these superconformal transformations, as we claimed. If we momentarily set $\alpha' = 1$ and if we consider only the holomorphic part, we get

$$
\begin{aligned}
\delta_\eta^{sc} S &= \frac{1}{2\pi}\int_{WS} d^2z\left[\frac{1}{\sqrt{2}}\partial(\eta\psi)\cdot\bar\partial X + \frac{1}{\sqrt{2}}\partial X\cdot\bar\partial(\eta\psi)\right.\\
&\qquad\qquad\qquad \left. +\frac{1}{2}\left(-\sqrt{2}\eta\partial X\right) + \frac{1}{2}\psi\cdot\partial\left(-\sqrt{2}\eta\partial X\right)\right]\\
&= \frac{1}{2\pi}\int_{WS} d^2z\left[\frac{2}{\sqrt{2}}\partial(\eta\psi)\cdot\bar\partial X - \sqrt{2}(\eta\partial X)\cdot\bar\partial\psi\right]\\
&\overset{IBP}{=} \frac{1}{2\pi}\int_{WS} d^2z\,\sqrt{2}\big[\partial(\eta\psi)\cdot\bar\partial X + \eta(\bar\partial\partial X)\cdot\psi\big]\\
&\overset{IBP}{=} \frac{1}{2\pi}\int_{WS} d^2z\,\sqrt{2}\big[\partial(\eta\psi)\cdot\bar\partial X - \bar\partial X\cdot\partial(\eta\psi)\big]\\
&= 0\,.
\end{aligned}
\tag{6.10}
$$

For the anti-holomorphic sector the reasoning is analogous and the result is the same. We hence proved that the worldsheet action (6.1) is invariant under superconformal transformations.

Let us now evaluate

$$
\begin{aligned}
\left[\delta_{\eta_1}^{sc}, \delta_{\eta_2}^{sc}\right] X^\mu &= \left(\delta_{\eta_1}^{sc}\delta_{\eta_2}^{sc} - \delta_{\eta_2}^{sc}\delta_{\eta_1}^{sc}\right) X^\mu \\
&= \delta_{\eta_1}^{sc}\left(\frac{1}{\sqrt{2}}\eta_2\psi^\mu\right) - \delta_{\eta_2}^{sc}\left(\frac{1}{\sqrt{2}}\eta_1\psi^\mu\right) \\
&= \frac{1}{\sqrt{2}}\eta_2(-\sqrt{2}\eta_1\partial X^\mu) + \frac{1}{\sqrt{2}}\eta_1\left(-\sqrt{2}\eta_2\partial X^\mu\right) \\
&= -\eta_2\eta_1\partial X^\mu + \eta_1\eta_2\partial X^\mu \\
&= 2\eta_1\eta_2\partial X^\mu \\
&= \delta_{2\eta_1\eta_2}^{c} X^\mu = \delta_{2\eta_2\eta_1}^{c} X^\mu,
\end{aligned}
\tag{6.11}
$$

$$
\begin{aligned}
\left[\delta_{\eta_1}^{sc}, \delta_{\eta_2}^{sc}\right] \psi^\mu &= \left(\delta_{\eta_1}^{sc}\delta_{\eta_2}^{sc} - \delta_{\eta_2}^{sc}\delta_{\eta_1}^{sc}\right) \psi^\mu \\
&= \delta_{\eta_1}^{sc}\left(-\sqrt{2}\eta_2\partial X^\mu\right) - \delta_{\eta_2}^{sc}\left(-\sqrt{2}\eta_1\partial X^\mu\right) \\
&= -\sqrt{2}\eta_2\partial\left(\sqrt{\frac{1}{2}}\eta_1\psi^\mu\right) + \sqrt{2}\eta_1\partial\left(\sqrt{\frac{1}{2}}\eta_2\psi^\mu\right) \\
&= -\eta_2\partial\left(\eta_1\psi^\mu\right) + \eta_1\partial\left(\eta_2\psi^\mu\right) \\
&= \left[-\eta_2\partial\eta_1 - (\partial\eta_2)\eta_1\right]\psi^\mu - 2\eta_2\eta_1\partial\psi^\mu \\
&= -\left[\frac{1}{2}\partial(2\eta_2\eta_1)\psi^\mu + 2\eta_2\eta_1(\partial\psi^\mu)\right] \\
&= \delta_{2\eta_2\eta_1}^{c}\psi^\mu.
\end{aligned}
\tag{6.12}
$$

We therefore just proved that

$$
\boxed{\left[\delta_{\eta_1}^{sc}, \delta_{\eta_2}^{sc}\right] = \delta_{2\eta_2\eta_1}^{c},}
\tag{6.13}
$$

which means that the composition of two superconformal transformations generates a conformal transformation. In a sense, we can say that superconformal transformations are the square root of conformal transformations.

As we already know (see eq. (4.45) and eq. (4.46)), conformal transformations are generated by the Noether charge

$$
T_{\epsilon(z)} = \oint_0 \frac{dz}{2\pi i}\epsilon(z)T^{(m)}(z),
\tag{6.14}
$$

such that

$$
\delta_\epsilon^{c}\phi(z) = -\left[T_\epsilon, \phi(z)\right].
\tag{6.15}
$$

Analogously, superconformal transformations are generated by the Noether charge

$$
G_{\eta(z)} = \oint_0 \frac{dz}{2\pi i}\eta(z)G^{(m)}(z),
\tag{6.16}
$$

such that

$$
\delta_\eta^{sc}\phi(z) = -\left[G_\eta, \phi(z)\right].
\tag{6.17}
$$

The current $G^{(\mathrm{m})}(z)$ is commonly called *supercurrent* and is defined as

$$G^{(\mathrm{m})}(z) = i\sqrt{\frac{2}{\alpha'}}\psi^\mu \partial X_\mu = \psi \cdot j\,, \tag{6.18}$$

where we recalled the definition (4.120a) of the bosonic current $j^\mu$.

As we previously saw (see eq. (4.56)), the composition (4.38) of two conformal transformations generates the correct OPE (4.66) of two stress-energy tensors. Analogously, eq. (6.13) fixes the contractions

$$\underline{G^{(\mathrm{m})}(z_1)G^{(\mathrm{m})}(z_2)} = \frac{d}{(z_1-z_2)^3} + \frac{2T^{(\mathrm{m})}(z_2)}{(z_1-z_2)}\,, \tag{6.19}$$

$$\underline{T^{(\mathrm{m})}(z_1)G^{(\mathrm{m})}(z_2)} = \frac{\frac{3}{2}G^{(\mathrm{m})}(z_2)}{(z_1-z_2)^2} + \frac{\partial G^{(\mathrm{m})}(z_2)}{(z_1-z_2)}\,. \tag{6.20}$$

To summarize, we can write

$$T^{(\mathrm{m})}(z_1)T^{(\mathrm{m})}(z_2) = \frac{\frac{c}{2}}{(z_1-z_2)^4} + \frac{2T^{(\mathrm{m})}(z_2)}{(z_1-z_2)^2} + \frac{\partial T^{(\mathrm{m})}(z_2)}{(z_1-z_2)} + (\mathrm{reg.})\,, \tag{6.21a}$$

$$G^{(\mathrm{m})}(z_1)G^{(\mathrm{m})}(z_2) = \frac{\frac{2}{3}c}{(z_1-z_2)^3} + \frac{2T^{(\mathrm{m})}(z_2)}{(z_1-z_2)} + (\mathrm{reg.})\,, \tag{6.21b}$$

$$T^{(\mathrm{m})}(z_1)G^{(\mathrm{m})}(z_2) = \frac{\frac{3}{2}G^{(\mathrm{m})}(z_2)}{(z_1-z_2)^2} + \frac{\partial G^{(\mathrm{m})}(z_2)}{(z_1-z_2)} + (\mathrm{reg.})\,, \tag{6.21c}$$

which completely define the superconformal algebra of our $(X;\psi)$-SCFT (the same holds for the anti-holomorphic sector). This is the simplest example of a $\mathcal{N}=1$ superconformal field theory. This algebra plays the same role that the Virasoro algebra played in the bosonic string: it is an algebra of constraints which should be imposed on the states of the theory.

We are now interested in the irreps of such an algebra, namely in the primary fields. As usual, a primary field $\phi^{(h)}(z)$ of weight $h$ is defined by

$$T^{(\mathrm{m})}(z_1)\phi^{(h)}(z_2) = \frac{h\phi^{(h)}(z_2)}{(z_1-z_2)^2} + \frac{\partial \phi^{(h)}(z_2)}{(z_1-z_2)} + (\text{regular terms})\,. \tag{6.22}$$

Therefore, because of eq. (6.21c), $G^{(\mathrm{m})}(z)$ is a primary field of weight $\frac{3}{2}$.

We can also define *superconformal primary* fields as follows. If $\phi^{(h)}(z)$ is a primary field of weight $h$, it is a superconformal primary (of weight $h$) if

$$G^{(\mathrm{m})}(z_1)\phi^{(h)}(z_2) = \frac{\psi^{(h+\frac{1}{2})}(z_2)}{(z_1-z_2)} + (\text{regular terms})\,. \tag{6.23}$$

This means that the supercurrent $G^{(\mathrm{m})}(z)$ acts on a superconformal primary field $\phi^{(h)}$ and it generates a superdescendant field $\psi^{(h+\frac{1}{2})}$.

In the $(X;\psi)$-SCFT we explicitly have

$$\underline{G^{(\mathrm{m})}(z_1)\psi^\mu(z_2)} = j \cdot \underline{\psi(z_1)\psi^\mu(z_2)} = \frac{j^\mu(z_1)}{(z_1-z_2)} \stackrel{\text{Taylor}}{=} \frac{j^\mu(z_2)}{(z_1-z_2)} + O(1)\,, \tag{6.24}$$

so that $\psi(z)$ is a superconformal primary field of weight $\frac{1}{2}$. Moreover

$$\underline{G^{(\mathrm{m})}(z_1)j^\mu(z_2)} = \psi \cdot \underline{j(z_1)j^\mu(z_2)} = \frac{\psi^\mu(z_1)}{(z_1-z_2)^2} \overset{\text{Taylor}}{=} \frac{\psi^\mu(z_2)}{(z_1-z_2)^2} + \frac{\partial\,\psi^\mu(z_2)}{(z_1-z_2)}\,, \tag{6.25}$$

and hence $j(z)$ is not a superconformal primary field. If we then take into account the $X^\mu$ field, we have that

$$\underline{G^{(\mathrm{m})}(z_1)i\sqrt{\frac{2}{\alpha'}}X^\mu(z_2)} = \frac{\psi^\mu(z_2)}{(z_1-z_2)}\,. \tag{6.26}$$

Therefore $X^\mu$ is (from this point of view) a superconformal primary field. However recall that it is not really a primary but a logarithmic field (recall eq. (4.130c)). On the other hand, if we consider a chiral plane wave we get

$$\begin{aligned}
\underline{G^{(\mathrm{m})}(z_1)\,{:}e^{iP\cdot X_L}{:}\,(z_2)} &= \psi\cdot\underline{j(z_1)\,{:}e^{iP\cdot X_L}{:}\,(z_2)}\\
&= \psi_\mu(z_1)iP_\nu\,\underline{j^\mu(z_1)X_L^\nu(z_2)}\,{:}e^{iP\cdot X_L}{:}\,(z_2)\\
&= \sqrt{\frac{\alpha'}{2}}P_\mu\psi^\mu\,{:}e^{iP\cdot X_L}{:}\,(z_2)\frac{1}{(z_1-z_2)}\,.
\end{aligned} \tag{6.27}$$

Therefore the plane wave ${:}e^{iP\cdot X_L}{:}$ is a superconformal primary of weight $\left(\frac{\alpha'P^2}{4},0\right)$.

### 6.1.2 Worldsheet supersymmetry

As we already know, translations are generated by $\partial_z = \partial$ (or $\partial_{\bar z} = \bar\partial$), namely by the adjoint action of $L_{-1}$:

$$[L_{-1},\cdot\,] = \oint_0 \frac{dz}{2\pi i}T^{(\mathrm{m})}(z)\,. \tag{6.28}$$

Indeed we have

$$[L_{-1},\phi(z_2)] = \oint_0 \frac{dz_1}{2\pi i}T^{(\mathrm{m})}(z_1)\phi(z_2) = \partial\,\phi(z_2)\,. \tag{6.29}$$

We now want to do the same with the supercurrent $G^{(\mathrm{m})}(z)$. We therefore define the holomorphic *supersymmetry* transformation (the anti-holomorphic case is analogous) as

$$\boxed{\delta_{\mathrm{SUSY}} = \oint_0 \frac{dz}{2\pi i}G^{(\mathrm{m})}(z)\,.} \tag{6.30}$$

The transformation $\delta_{\mathrm{SUSY}}$ is Grassmann odd, since $G^{(\mathrm{m})}(z)$ is Grassmann odd. If we then recall that $T^{(\mathrm{m})}(z)$ can be expanded in terms of Virasoro operators $L_n$ (see eq. (4.44)), we can expand the supercurrent $G^{(\mathrm{m})}(z)$ as

$$G^{(\mathrm{m})}(z) = \sum_{r\in\mathbb{Z}+\frac{1}{2}} G_r z^{-r-\frac{3}{2}}\,. \tag{6.31}$$

Thus we have that (with an understood graded commutator)

$$\delta_{\mathrm{SUSY}} = \left[G_{-\frac{1}{2}},\cdot\,\right]\,. \tag{6.32}$$

The action of the supersymmetry on the fields of our $(X;\psi)$-SCFT is

$$\delta_{\text{SUSY}}\left(i\sqrt{\frac{2}{\alpha'}}X^{\mu}(z)\right) = \psi^{\mu}(z)\,, \tag{6.33a}$$

$$\delta_{\text{SUSY}}(\psi^{\mu}(z)) = j^{\mu}(z)\,, \tag{6.33b}$$

$$\delta_{\text{SUSY}}(j^{\mu}(z)) = \partial\psi^{\mu}(z)\,. \tag{6.33c}$$

Two fields connected by a supersymmetry transformations are called *superpartners*. Moreover, the supersymmetry is the "square root" of translations:

$$[\delta_{\text{SUSY}}, \delta_{\text{SUSY}}] = \partial_{z}\,. \tag{6.34}$$

**Exercise 6.1.2**

Prove eq. (6.34).

### 6.1.3  Ghost $(b,c;\beta,\gamma)$-SCFT

As we saw previously, to the matter $X$-CFT we associate a ghost $(b,c)$-CFT through the Faddeev-Popov mechanism. In the same way, when we add the matter $\psi$-CFT, we have to add by WS supersymmetry a ghost $(\beta,\gamma)$-CFT. The complete ghost CFT is given by the action

$$S_{\text{gh}} = \frac{1}{2\pi}\int_{\text{WS}}d^{2}z\left(b\bar{\partial}c + \bar{b}\partial\bar{c} + \beta\bar{\partial}\gamma + \bar{\beta}\partial\bar{\gamma}\right), \tag{6.35}$$

where

- $b$ is a primary of weight 2 and, being the FP ghost of diff×Weyl symmetry, it is Grassmann odd, namely a fermion (and hence nilpotent),

- $c$ is a primary of weight $-1$ and, being the FP ghost of diff×Weyl symmetry, it is Grassmann odd, namely a fermion (and hence nilpotent),

- $\beta$ is a primary of weight $\frac{3}{2}$ and, being the FP ghost of a worldsheet fermionic symmetry, it is Grassmann even, namely a boson,

- $\gamma$ is a primary of weight $-\frac{1}{2}$ and, being the FP ghost of a worldsheet fermionic symmetry, it is Grassmann even, namely a boson.

The OPEs are

$$b(z_1)c(z_2) = \frac{1}{(z_1-z_2)} + (\text{regular terms})\,, \tag{6.36a}$$

$$c(z_1)b(z_2) = \frac{1}{(z_1-z_2)} + (\text{regular terms})\,, \tag{6.36b}$$

$$\beta(z_1)\gamma(z_2) = -\frac{1}{(z_1-z_2)} + (\text{regular terms})\,, \tag{6.36c}$$

$$\gamma(z_1)\beta(z_2) = \frac{1}{(z_1-z_2)} + (\text{regular terms})\,. \tag{6.36d}$$

The (holomorphic) stress energy tensor is defined as

$$T^{(\mathrm{gh})}(z) = T^{(b,c)}(z) + T^{(\beta,\gamma)}(z), \tag{6.37}$$

where

$$T^{(b,c)}(z) = \ :(\partial b)c:(z) - 2\ :\partial(bc):(z), \tag{6.38a}$$

$$T^{(\beta,\gamma)}(z) = \ :(\partial \beta)\gamma:(z) - \frac{3}{2}\ :\partial(\beta\gamma):(z). \tag{6.38b}$$

> **Exercise 6.1.3**
>
> Using what we said so far, prove that $\beta$ is a primary of weight $\frac{3}{2}$ and that $\gamma$ is a primary of weight $-\frac{1}{2}$.

Moreover, the leading terms of the OPEs

$$T^{(b,c)}(z_1)T^{(b,c)}(z_2) \approx \frac{\frac{1}{2}(-26)}{(z_1 - z_2)^4}, \tag{6.39a}$$

$$T^{(\beta,\gamma)}(z_1)T^{(\beta,\gamma)}(z_2) \approx \frac{\frac{1}{2}(11)}{(z_1 - z_2)^4}, \tag{6.39b}$$

tell us that

$$c^{(b,c)} = -26, \tag{6.40a}$$

$$c^{(\beta,\gamma)} = 11, \tag{6.40b}$$

which means that

$$c^{(\mathrm{gh})} = c^{(b,c)} + c^{(\beta,\gamma)} = -15. \tag{6.41}$$

> **Exercise 6.1.4**
>
> Prove eq. (6.39b).

The $(b,c\,;\beta,\gamma)$-SCFT superalgebra is based on the ghost supercurrent

$$G^{(\mathrm{gh})}(z) = -\frac{1}{2}(\partial\beta)c + \frac{3}{2}\partial(\beta c) - 2b\gamma, \tag{6.42}$$

where the normal ordering is not needed since the $(b,c)$ and the $(\beta,\gamma)$ systems do not talk to each other. One can then show that $T^{(\mathrm{gh})}$ and $G^{(\mathrm{gh})}$ generate a superalgebra of central charge $c^{(\mathrm{gh})} = -15$.

If we now put together the matter central charge and the ghost one, we observe that the total anomaly

$$c^{(\mathrm{tot})} = c^{(\mathrm{m})} + c^{(\mathrm{gh})} = \frac{3}{2}d - 15, \tag{6.43}$$

vanishes when

$$d = 10. \tag{6.44}$$

This is the *critical dimension* of the superstring (for the bosonic string, it was $d = 26$).

Finally, we can recall that in the bosonic string the $TT$ algebra was interpreted as a constraint algebra. The missing EOM for the metric, namely $T^{(\mathrm{m})} = 0$, was the classical constraint.

Then BRST quantization added to each constraint a ghost (with an additional term, due to the non-Abelian nature of the constraints), hence defining the current

$$j_B(z) = c\,T^{(m)}(z) + \frac{1}{2} : c\,T^{(gh)} : (z). \tag{6.45}$$

This led to the definition of the BRST charge

$$Q_B = \oint_0 \frac{dz}{2\pi i} j_B(z), \tag{6.46}$$

which is nilpotent iff the total central charge of the algebra vanishes:

$$Q_B^2 = 0 \iff c^{(m)} + c^{(gh)} = 0. \tag{6.47}$$

Analogously, in the superstring case we have two sets of classical constraints, given by

$$T^{(m)} = 0, \tag{6.48a}$$
$$G^{(m)} = 0, \tag{6.48b}$$

where the first equation is again the missing EOM for the metric, while the second one (as we will later argue) is the missing EOM for the superpartner of the metric (the so-called gravitino). If we then associate the $(b, c)$-system to $T^{(m)}$ and the $(\beta, \gamma)$-system to $G^{(m)}$, the BRST current is constructed as

$$j_B(z) = c\,T^{(m)}(z) + \gamma\,G^{(m)}(z) + \frac{1}{2}\left(:c\,T^{(gh)}:(z) + :\gamma\,G^{(gh)}:(z)\right), \tag{6.49}$$

and the BRST charge is, as usual, the integral of such a current:

$$Q_B = \oint_0 \frac{dz}{2\pi i} j_B(z). \tag{6.50}$$

Again we have that

$$Q_B^2 = 0 \iff c^{(m)} + c^{(gh)} = 0, \tag{6.51}$$

which is satisfied at the critical dimension $d = 10$.

Moreover, one can show that

$$[Q_B, b(z)] = T^{(tot)}(z) = T^{(m)}(z) + T^{(gh)}(z), \tag{6.52a}$$
$$[Q_B, \beta(z)] = G^{(tot)}(z) = G^{(m)}(z) + G^{(gh)}(z). \tag{6.52b}$$

**Exercise 6.1.5**

Prove eqs. (6.52).

In conclusion, all the fields of our superconformal theory are related by WS supersymmetry

$$\psi^\mu\left(h = \frac{1}{2}\right) \quad \longleftrightarrow \quad X^\mu\ (h = 0), \tag{6.53a}$$

$$\gamma\left(h = -\frac{1}{2}\right) \quad \longleftrightarrow \quad c\ (h = -1), \tag{6.53b}$$

$$\beta\left(h = \frac{3}{2}\right) \quad \longleftrightarrow \quad b\ (h = 2). \tag{6.53c}$$

## 6.2 Closed superstring

### 6.2.1 Neveu-Schwarz and Ramond sectors

Let us now consider a closed superstring. The worldsheet spinors we introduced (that we will generically denote as $\chi$, i.e. $\chi = \psi, \beta, \gamma$) can be periodic or anti-periodic along the $\sigma$ direction:

- periodic: $\chi(\sigma + 2\pi) = +\chi(\sigma)$,

- anti-periodic: $\chi(\sigma + 2\pi) = -\chi(\sigma)$.

These two conditions are both possible, because the density given by spinor bilinears remains the same. On the other hand, the anti-periodic condition would not be acceptable in the bosonic $X$ case, since it would break Lorentz invariance.[34]

If we now write as usual $w = t + i\sigma$, we can define two sectors:

$$\chi(w + 2\pi i) = \begin{cases} +\chi(w), & \text{Ramond (R) sector,} \\ -\chi(w), & \text{Neveu-Schwarz (NS) sector.} \end{cases} \tag{6.54}$$

Analogously, since the supercurrent is a fermion, we can write

$$G(w + 2\pi i) = \begin{cases} +G(w) & \text{(R)}, \\ -G(w) & \text{(NS)}. \end{cases} \tag{6.55}$$

If we then map the WS cylinder on the complex plane via the exponential map $z = e^w$, recalling that the conformal weight of $\psi$ is $\frac{1}{2}$ we can write

$$\psi(z)(dz)^{\frac{1}{2}} = \psi(w)(dw)^{\frac{1}{2}}, \tag{6.56}$$

and hence

$$\begin{aligned} \psi(z) &= \left(\frac{dz}{dw}\right)^{\frac{1}{2}} \psi(w(z)) \\ &= \left(\frac{d(e^w)}{dw}\right)^{\frac{1}{2}} \psi(\log(z)) \\ &= \frac{1}{\sqrt{z}} \psi(\log(z)). \end{aligned} \tag{6.57}$$

Therefore we can write the oscillator expansion of the $\psi$ field in its two possible sectors:

$$\begin{cases} \psi_{\text{NS}}^{\mu}(z) = \sum_{r \in \mathbb{Z} + \frac{1}{2}} \psi_r^{\mu} z^{-r-\frac{1}{2}}, \\ \psi_{\text{R}}^{\mu}(z) = \sum_{n \in \mathbb{Z}} \psi_n^{\mu} z^{-n-\frac{1}{2}}. \end{cases} \tag{6.58}$$

In the NS sector there are no branch cuts on $\mathbb{C}$ (see fig. 6.16.1(a)), since for $r \in \mathbb{Z} + \frac{1}{2}$ the function $z^{-r-\frac{1}{2}}$ is meromorphic (because $-r - \frac{1}{2} \in \mathbb{Z}$). In the R sector, on the other hand, there is a branch cut from 0 to $\infty$ (see fig. 6.16.1(b)) because for $n \in \mathbb{Z}$ the function $z^{-n-\frac{1}{2}}$ is double-valued (since $-n - \frac{1}{2} \in \mathbb{Z} + \frac{1}{2}$).

---

[34]However this possibility is still considered in string theory in the construction of *orbifolds*.

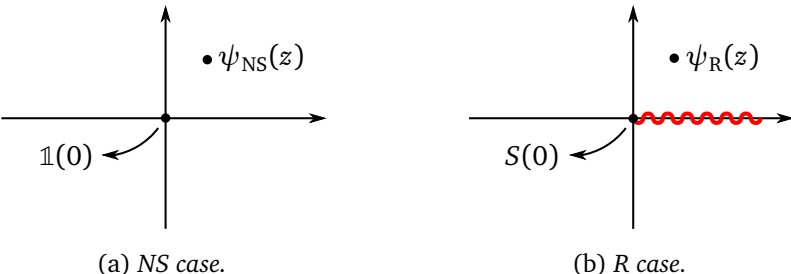

(a) *NS case.*  (b) *R case.*

Figure 6.1: The fermion $\psi(z)$ in the NS and R cases. In the NS plane there are no branch cuts and in the origin there is the identity field, corresponding to the $SL(2, C)$ vacuum. On the other hand, in the R case there is a branch cut (marked in red) due to the spin field inserted in the origin, which defines a new vacuum, the R vacuum.

The presence of the branch cut can be understood as an insertion of a peculiar field in the origin of the complex plane, which changes the nature of the $SL(2, C)$ vacuum. This field $S(0)$ is called *spin field* and its contraction with a fermion $\psi$ behaves as a square root:

$$\underline{\psi(z_1)S(z_2)} = \frac{(\cdots)}{(z_1 - z_2)^{\pm\frac{1}{2}}} \, . \tag{6.59}$$

The OPE of two fermions $\psi$ remains the same in both sectors:

$$\psi(z_1)\psi(z_2) = \frac{1}{(z_1 - z_2)} + (\text{regular terms}), \tag{6.60}$$

but it is realized as two different oscillator algebras

$$\text{NS)} \quad \left[\psi_r^\mu, \psi_s^\nu\right] = \eta^{\mu\nu}\delta_{r+s,0} \, , \quad \text{with } r, s \in \mathbb{Z} + \frac{1}{2} \, , \tag{6.61a}$$

$$\text{R)} \quad \left[\psi_n^\mu, \psi_m^\nu\right] = \eta^{\mu\nu}\delta_{n+m,0} \, , \quad \text{with } n, m \in \mathbb{Z} \, . \tag{6.61b}$$

The same can be said for $(\beta, \gamma)$ ghosts. We have that

$$\begin{cases} \beta_{\text{NS}}(z) = \sum_{r\in\mathbb{Z}+\frac{1}{2}} \beta_r z^{-r-\frac{3}{2}} \, , \\ \beta_{\text{R}}(z) = \sum_{n\in\mathbb{Z}} \beta_n z^{-n-\frac{3}{2}} \, , \end{cases} \tag{6.62a}$$

$$\begin{cases} \gamma_{\text{NS}}(z) = \sum_{r\in\mathbb{Z}+\frac{1}{2}} \gamma_r z^{-r+\frac{1}{2}} \, , \\ \gamma_{\text{R}}(z) = \sum_{n\in\mathbb{Z}} \gamma_n z^{-n+\frac{1}{2}} \, , \end{cases} \tag{6.62b}$$

which implies

$$\text{NS)} \quad [\beta_r, \gamma_s] = \delta_{r+s,0} \, , \quad \text{with } r, s \in \mathbb{Z} + \frac{1}{2} \, , \tag{6.63a}$$

$$\text{R)} \quad [\beta_n, \gamma_m] = \delta_{n+m,0} \, , \quad \text{with } n, m \in \mathbb{Z} \, . \tag{6.63b}$$

Analogously, for the supercurrent we can write

$$\begin{cases} G_{\text{NS}}(z) = \sum_{r\in\mathbb{Z}+\frac{1}{2}} G_r z^{-r-\frac{3}{2}} \, , \\ G_{\text{R}}(z) = \sum_{n\in\mathbb{Z}} G_n z^{-n-\frac{3}{2}} \, , \end{cases} \tag{6.64}$$

which implies

$$\text{NS)} \quad [G_r, G_s] = 2L_{r+s} + \frac{c}{12}(4r^2 - 1)\delta_{r+s,0} \, , \quad \text{with } r, s \in \mathbb{Z} + \frac{1}{2} \, , \tag{6.65a}$$

$$\text{R)} \quad [G_n, G_m] = 2L_{n+m} + \frac{c}{12}(4n^2 - 1)\delta_{n+m,0} \, , \quad \text{with } n, m \in \mathbb{Z} \, . \tag{6.65b}$$

Therefore the superconformal algebra can be written in terms of oscillators as

$$[L_n, L_m] = (n-m)L_{n+m} + \frac{c}{12}n(n^2-1)\delta_{n+m,0}, \tag{6.66a}$$

$$[L_n, G_r] = \frac{n-2r}{2}G_{n+r}, \tag{6.66b}$$

$$\text{NS)} \quad [G_r, G_s] = 2L_{r+s} + \frac{c}{12}(4r^2-1)\delta_{r+s,0}, \quad \text{with } r,s \in \mathbb{Z}+\frac{1}{2}, \tag{6.66c}$$

$$\text{R)} \quad [G_r, G_s] = 2L_{r+s} + \frac{c}{12}(4r^2-1)\delta_{r+s,0}, \quad \text{with } r,s \in \mathbb{Z}, \tag{6.66d}$$

where eq. (6.66b) comes from the fact that $G$ is a primary field.

This algebra is generated by two SCFTs:

- matter $(X;\psi)$-SCFT with central charge $c^{(\text{m})} = \frac{3}{2}d$,

- ghost $(b,c;\beta,\gamma)$-SCFT with central charge $c^{(\text{gh})} = -15$.

As already said above, the total SCFT has $c^{(\text{tot})} = c^{(\text{m})} + c^{(\text{gh})} = 0$ when $d = 10$.

### 6.2.2 Neveu-Schwarz and Ramond vacua

Let us then built the vacua of the theory. We will use the following notation

- $\alpha_n^\mu$: oscillators in the $X$-CFT,

- $\psi_r^\mu$: oscillators in the $\psi_{\text{NS}}$-CFT,

- $\psi_n^\mu$: oscillators in the $\psi_{\text{R}}$-CFT.

We then have two possibilities.

- In the NS sector the definite-momentum vacuum $|0,P\rangle_{\text{NS}}$ is such that

$$\alpha_n^\mu |0,P\rangle_{\text{NS}} = 0, \quad \text{for } n \geq 1, \tag{6.67a}$$

$$\psi_r^\mu |0,P\rangle_{\text{NS}} = 0, \quad \text{for } r \geq \frac{1}{2}. \tag{6.67b}$$

- In the R sector the definite-momentum vacuum $|0,P\rangle_{\text{R}}$ is such that

$$\alpha_n^\mu |0,P\rangle_{\text{R}} = 0, \quad \text{for } n \geq 1, \tag{6.68a}$$

$$\psi_n^\mu |0,P\rangle_{\text{R}} = 0, \quad \text{for } n \geq 1. \tag{6.68b}$$

Let us now recall that $\left[\psi_n^\mu, \psi_m^\nu\right] = \eta^{\mu\nu}\delta_{n+m,0}$. This implies that for the zero-modes we have

$$\left[\psi_0^\mu, \psi_0^\nu\right] = \eta^{\mu\nu}. \tag{6.69}$$

Therefore, just like $b_0$ and $c_0$, the zero-mode $\psi_0^\mu$ is neither a creation nor an annihilation operator.

Since we have many of these zero-modes, the Ramond vacuum is degenerate. The degeneracy is labelled by a label $\alpha$. Then we can write

$$\psi_0^\mu |\alpha,P\rangle_{\text{R}} = (\psi^\mu)_{\alpha\beta} |\beta,P\rangle_{\text{R}}. \tag{6.70}$$

If we now define

$$\Gamma^\mu = \sqrt{2}\psi_0^\mu, \tag{6.71}$$

from eq. (6.69) we recognize the Clifford algebra in $d$ dimensions:

$$[\Gamma^\mu, \Gamma^\nu] = 2\eta^{\mu\nu}, \tag{6.72}$$

where obviously the square brackets stand for an anti-commutator, since we are dealing with fermionic objects (recall that we are using the graded commutator convention). This means that the Ramond vacuum is the space where gamma matrices (of dimension $d$) act, namely the spinor space in $d$ dimensions (see appendix A). Therefore, if we work in the critical dimension $d = 10$, the $\psi_R$ sector Ramond vacuum $|\alpha\rangle_R$ is a 10-dimensional spinor of $SO(1,9)$, and it can be decomposed in two irreps as

$$|\alpha\rangle_R = \underbrace{|\alpha\rangle}_{16_c} \oplus \underbrace{|\dot\alpha\rangle}_{16_s}, \tag{6.73}$$

where $16_c$ is the positive chirality 16-dimensional irrep, while $16_s$ is the negative chirality 16-dimensional irrep. These two irreps are defined by the action of the chirality matrix $\Gamma$ as

$$\Gamma |\alpha\rangle = +|\alpha\rangle, \tag{6.74a}$$
$$\Gamma |\dot\alpha\rangle = -|\dot\alpha\rangle. \tag{6.74b}$$

Given what we said so far, recalling that the momentum is given by the definite-momentum vacuum in the $X$ sector, we can write the total matter vacuum in the NS sector as

$$|0, P\rangle_{NS}^{matter} = \underbrace{|0, P\rangle}_{X \text{ sector}} \otimes \underbrace{|0\rangle_{NS}}_{\psi \text{ sector}} = |0, P\rangle \otimes |0\rangle, \tag{6.75}$$

where we wrote $|0\rangle_{NS} = |0\rangle$ since, at the matter level, the NS vacuum in the $\psi$ sector is precisely the $SL(2,\mathbb{C})$-invariant vacuum.

On the other hand, the total vacuum in the R sector can be written as

$$|\alpha, P\rangle_R^{matter} = \underbrace{|0, P\rangle}_{X \text{ sector}} \otimes \underbrace{|\alpha\rangle_R}_{\psi \text{ sector}} = |0, P\rangle \otimes \left( |\alpha\rangle \oplus |\dot\alpha\rangle \right). \tag{6.76}$$

As we anticipated, the R vacuum is obtained by inserting a spin field to the (NS) $SL(2,\mathbb{C})$-invariant vacuum:

$$|\overset{(\cdot)}{\alpha}\rangle = S_{\overset{(\cdot)}{\alpha}}(0)|0\rangle. \tag{6.77}$$

Therefore the Ramond vacuum is in a sense an excited state of the Neveu-Schwarz vacuum. This is analogous to what we found previously in the $X$-BCFT for the vacuum of open strings satisfying mixed Neumann-Dirichlet boundary conditions.

### 6.2.3 Old covariant quantization

#### 6.2.3.1 Constraints

We are now able to write every possible state in the holomorphic matter sector (for the anti-holomorphic sector everything works in the same way). The most generic state is made up of a polarization tensor, $X$ oscillators (bosonic sector) and $\psi$ oscillators (fermionic sector) acting on the definite momentum vacuum.

In the NS sector we have

$$G_{\{\mu_1\cdots\mu_k\}\{\nu_1\cdots\nu_q\}}(P)\left(\alpha_{-n_1}^{\mu_1}\cdots\alpha_{-n_k}^{\mu_k}\right)\left(\psi_{-r_1}^{\nu_1}\cdots\psi_{-r_q}^{\nu_q}\right)|0,P\rangle_{\text{NS}}\,, \tag{6.78}$$

which is the most generic Lorentz representation for an integer-spin field, namely for a spacetime boson. Indeed $\{\mu_1\ldots\mu_k\}$ and $\{\nu_1\ldots\nu_q\}$ are vectorial Lorentz indices.

On the other hand, in the R sector we have

$$\chi_{\alpha\{\mu_1\cdots\mu_k\}\{\nu_1\cdots\nu_q\}}(P)\left(\alpha_{-n_1}^{\mu_1}\cdots\alpha_{-n_k}^{\mu_k}\right)\left(\psi_{-r_1}^{\nu_1}\cdots\psi_{-r_q}^{\nu_q}\right)|\alpha,P\rangle_{\text{R}}\,, \tag{6.79}$$

which is the most generic Lorentz representation for a half-integer-spin field, namely for a spacetime fermion. Indeed $\alpha$ is a spinorial index in $d=10$, while $\{\mu_1\ldots\mu_k\}$ and $\{\nu_1\ldots\nu_q\}$ are again vectorial Lorentz indices. This means that we finally found fermions in string theory. But everything is yet to be set on-shell, we still have to impose the physical constraints.

Let us first look at the old covariant quantization. For $n\in\mathbb{Z}$ and $r\in\mathbb{Z}+\frac{1}{2}$, the constraints are

- in the NS sector:

$$(L_0-a_{\text{NS}})\,|\text{phys}\rangle_{\text{NS}}=0\,, \tag{6.80a}$$

$$L_{n>0}\,|\text{phys}\rangle_{\text{NS}}=0\,, \tag{6.80b}$$

$$G_{r\geq\frac{1}{2}}\,|\text{phys}\rangle_{\text{NS}}=0\,, \tag{6.80c}$$

- in the R sector:

$$(L_0-a_{\text{R}})\,|\text{phys}\rangle_{\text{R}}=0\,, \tag{6.81a}$$

$$L_{n>0}\,|\text{phys}\rangle_{\text{R}}=0\,, \tag{6.81b}$$

$$G_{n\geq0}\,|\text{phys}\rangle_{\text{R}}=0\,. \tag{6.81c}$$

If we then incorporate the ghost contribution, from BRST (or lightcone) quantization we have that (no proof at this stage)

$$\boxed{a_{\text{NS}}=\frac{1}{2}\,,\qquad a_{\text{R}}=\frac{5}{8}\,.} \tag{6.82}$$

In order to understand the origin of these values of the normal ordering constant $a$, we may recall that, in the bosonic string, $a$ was the conformal weight of the ghost vacuum $c_1|0\rangle$:

$$\begin{aligned}L_0^{(\text{tot})}&=L_0^{(\text{m})}+L_0^{(\text{gh})}\\&=L_0^{(\text{m})}-a\\&=L_0^{(\text{m})}-1\\&=0\,.\end{aligned} \tag{6.83}$$

Now, coming back to the superstring, it can be shown that the matter+ghost NS vacuum is related to the $SL(2,\mathbb{C})$-invariant vacuum as

$$c_1 |0\rangle_{\text{NS}} = c\mathcal{V}^{(\beta,\gamma)}_{\frac{1}{2}}(0)|0\rangle\,, \tag{6.84}$$

where $\mathcal{V}^{(\beta,\gamma)}_{1/2}$ is a primary operator of weight $1/2$ in the $(\beta,\gamma)$-SCFT, which can be written as $\mathcal{V}^{(\beta,\gamma)}_{1/2} = e^{-\varphi}$ (see 6.5.1 for more details on this topic). It is hence clear that $a_{\text{NS}} = 1/2$ is the weight of the NS vacuum, since the total weight of $c\mathcal{V}^{(\beta,\gamma)}_{1/2}$ is $1-1/2 = -1/2$. Therefore in the NS sector we have that

$$\begin{aligned} L_0^{(\text{tot})} &= L_0^{(\text{m})} + L_0^{(\text{gh})} \\ &= L_0^{(\text{m})} - a_{\text{NS}} \\ &= L_0^{(\text{m})} - \frac{1}{2} \\ &= 0\,. \end{aligned} \tag{6.85}$$

On the other hand, in the R sector the matter+ghost vacuum can be written as

$$c_1 |\overset{(\cdot)}{\alpha}\rangle_{\text{R}} = S_{(\cdot)\atop\alpha} c\mathcal{V}^{(\beta,\gamma)}_{\frac{3}{8}}(0)|0\rangle\,, \tag{6.86}$$

where $\mathcal{V}^{(\beta,\gamma)}_{3/8}$ is a primary operator of weight $3/8$ in the $(\beta,\gamma)$-SCFT, which can be written as $\mathcal{V}^{(\beta,\gamma)}_{3/8} = e^{-\varphi/2}$ (this field $\varphi$ is the same we encountered in the NS case and it will be explained in 6.5.1). Therefore in the R sector we have that

$$\begin{aligned} L_0^{(\text{tot})} &= L_0^{(\text{m})} + L_0^{(\text{gh})} \\ &= L_0^{(\text{m})} - a_{\text{R}} \\ &= L_0^{(\text{m})} + \left(-1 + \frac{3}{8}\right) \\ &= L_0^{(\text{m})} - \frac{5}{8} \\ &= 0\,. \end{aligned} \tag{6.87}$$

Given what we said so far, we can write

$$\text{NS)} \quad L_n = \frac{1}{2}\sum_{k\in\mathbb{Z}} \alpha_{n-k}\cdot\alpha_k + \frac{1}{4}\sum_{r\in\mathbb{Z}+\frac{1}{2}} (2r-n):\psi_{n-r}\cdot\psi_r:\,, \tag{6.88a}$$

$$\text{R)} \quad L_n = \frac{1}{2}\sum_{k\in\mathbb{Z}} \alpha_{n-k}\cdot\alpha_k + \frac{1}{4}\sum_{k\in\mathbb{Z}} (2k-n):\psi_{n-k}\cdot\psi_k: + \frac{d}{16}\delta_{n,0}\,. \tag{6.88b}$$

We can notice that the R sector has a shift equal to the spin field weight, namely of $d/16$ (which in $d = 10$ is exactly $5/8$; see 6.5.1 for more details). It shifts the vacuum from the NS sector to the R sector.

Moreover we have that the supercurrent is defined as

$$G_r = \sum_{n\in\mathbb{Z}} \alpha_n\cdot\psi_{n-r}\,, \tag{6.89}$$

where $r\in\mathbb{Z}+1/2$ in the NS case, while $r\in\mathbb{Z}$ in the R case.

#### 6.2.3.2  Spectrum

Let us now write

$$L_0 = \frac{1}{2}(\alpha_0)^2 + N, \tag{6.90}$$

where $\alpha_0^\mu = \sqrt{\frac{\alpha'}{2}}P^\mu$ is the $X$-CFT zero-mode (in the closed string case, which is the one we are studying) and where $N$ is the level operator, defined in $d = 10$ as

$$\text{NS)} \quad N_{\text{NS}} = \frac{1}{2}\sum_{k\in\mathbb{Z}}\alpha_{-k}\cdot\alpha_k + \frac{1}{4}\sum_{r\in\mathbb{Z}+\frac{1}{2}}(2r):\psi_{-r}\cdot\psi_r:, \tag{6.91a}$$

$$\text{R)} \quad N_{\text{R}} = \frac{1}{2}\sum_{k\in\mathbb{Z}}\alpha_{-k}\cdot\alpha_k + \frac{1}{4}\sum_{k\in\mathbb{Z}}(2k):\psi_{-k}\cdot\psi_k: +\frac{5}{8}. \tag{6.91b}$$

We can then analyze the first levels of the spectrum in the NS sector.

■ **Level $N_{\text{NS}} = 0$**
At the level $N_{\text{NS}} = 0$ we have the tachyon

$$t(P)|0,P\rangle_{\text{NS}}, \tag{6.92}$$

which is lighter then the closed bosonic string tachyon because

$$L_0^{(\text{m})} - a_{\text{NS}} = L_0^{(\text{m})} - \frac{1}{2} = 0 \implies \frac{\alpha'P^2}{4} = \frac{1}{2} = a_{\text{NS}} \implies m^2 = -\frac{2}{\alpha'}. \tag{6.93}$$

■ **Level $N_{\text{NS}} = \frac{1}{2}$**
At the level $N_{\text{NS}} = 1/2$ the generic state is

$$\xi_\mu(P)\psi_{\frac{1}{2}}^\mu|0,P\rangle_{\text{NS}}, \tag{6.94}$$

which is a massless vector:

$$m^2 = \frac{4}{\alpha'}\left(\frac{1}{2} - a_{\text{NS}}\right) = 0. \tag{6.95}$$

If we look at the supercurrent, we have that

$$G_{\frac{1}{2}} = \sum_n \alpha_n \cdot \psi_{\frac{1}{2}-n}, \tag{6.96}$$

that for $n = 0$ becomes $\alpha_0 \cdot \psi_{\frac{1}{2}}$. Then the constraint (6.80c) is

$$G_{\frac{1}{2}} = 0 \implies \alpha_0 \cdot \xi = 0 \implies P \cdot \xi = 0, \tag{6.97}$$

where (as we usually do for $L_n$) we used the same notation for the operator $G_{1/2}$ and its eigenvalue on the physical state. If we take $a_{NS} = 1/2$ we find a transverse massless vector which falls in vector representation of the little group $SO(8)$, which we denote $8_V$. If we look at higher levels, we find massive states.

Let us now analyze the first level of the spectrum in the R sector.

Table 6.1: The first spectrum levels of the holomorphic sector of the closed superstring.

|  | NS | R |
| --- | --- | --- |
| $N_{\text{NS}} = 0:$   tachyon<br>$N_{\text{NS}} = \frac{1}{2}:$   $8_V$ massless | | $N_{\text{R}} = 0:$   $8_C \oplus 8_S$ massless |

■ **Level $N_{\text{R}} = 0$**

At the level $N_{\text{R}} = 0$ the generic state is

$$u_\alpha(P)|\alpha, P\rangle_{\text{R}} + v_{\dot\alpha}(P)|\dot\alpha, P\rangle_{\text{R}}, \tag{6.98}$$

where $u_\alpha(P)$ is a left spinor in $d = 10$, while $v_{\dot\alpha}(P)$ is a right spinor in $d = 10$. We are hence treating separately the two chirality irreps.

If we look at the supercurrent, we have that

$$G_0 = \sum_n \alpha_n \cdot \psi_{-n}, \tag{6.99}$$

on the R vacuum only $n = 0$ gives contribution

$$G_0 = \alpha_0 \cdot \psi_0 = \sqrt{\frac{\alpha'}{2}} P \cdot \Gamma \frac{1}{\sqrt{2}} = \frac{\sqrt{\alpha'}}{2} \not{P}, \tag{6.100}$$

where $\not{P}$ is the Dirac kinetic operator of spacetime fermions.

The constraint (6.80c) reads as (recalling that $(\not{P})^2 = P^2$)

$$\not{P}u(P) = 0 \implies m^2 = 0, \tag{6.101a}$$
$$\not{P}v(P) = 0 \implies m^2 = 0. \tag{6.101b}$$

We hence found the Dirac equation. As usual this equation reduces by half the spinorial d.o.f. Therefore the two states $u_\alpha(P)|\alpha, P\rangle_{\text{R}}$ and $v_{\dot\alpha}(P)|\dot\alpha, P\rangle_{\text{R}}$ fall into massless $SO(8)$ spinorial representations of dimension $2^{8/2} = 16 = 8 + 8$:

$$SO(8) \longrightarrow \underbrace{8_C}_{\text{left}} \oplus \underbrace{8_S}_{\text{right}}. \tag{6.102}$$

If we look at higher levels, we find massive states with half-integer spin.

What we found in the holomorphic sector is summed up in table 6.1. We notice that the massless part of the spectrum is organized in representations of $SO(8)$.

### 6.2.4   Lightcone quantization

As with the bosonic string, we can handle directly all the physical states by going to the lightcone gauge.

#### 6.2.4.1   Constraints

In the bosonic string we fixed the conformal transformations by writing

$$X_L^+(z) = \frac{X_0^+}{2} - i\alpha' P^+ \log z \qquad \text{(no oscillators)}, \tag{6.103}$$

and hence

$$j^+(z) = i\sqrt{\frac{2}{\alpha'}}\partial X^+(z) = \sqrt{\frac{\alpha'}{2}}P^+\frac{1}{z}. \tag{6.104}$$

In the superstring, however, we also have the $\psi$ sector and superconformal transformations. If we write in the lightcone direction $+$ the superconformal transformation

$$\delta_\eta^{\text{sc}}\psi^+(z) = -i\eta j^+(z), \tag{6.105}$$

we can choose to fix

$$\psi^+(z) = 0. \tag{6.106}$$

Therefore, in the superstring case, the lightcone gauge is

$$\begin{cases} X_L^+(z) = \frac{X_0^+}{2} - i\alpha'P^+\log z, \\ \psi^+(z) = 0. \end{cases} \tag{6.107}$$

At the classical level, we can implement the constraints $L_{n>0}|\text{phys}\rangle_{\text{NS}} = 0$ and $G_{r\geq\frac{1}{2}}|\text{phys}\rangle_{\text{NS}} = 0$ in the NS sector (recall eqs. (6.80)), or $L_{n>0}|\text{phys}\rangle_{\text{R}} = 0$ and $G_{n\geq 0}|\text{phys}\rangle_{\text{R}} = 0$ in the R sector (recall eqs. (6.81)). For the time being, we ignored the $L_0$ constraint. We then find that oscillators in the $-$ direction can be expressed in terms of transverse oscillators as

$$\begin{cases} \partial X_L^-(z) = \frac{z}{2P^+}\left(\frac{2}{\alpha'}\partial X^j\partial X_j + i\psi^j\partial\psi_j\right), \\ \psi^-(z) = \frac{2z}{\alpha'P^+}\psi^j\partial X_{L,j}, \end{cases} \tag{6.108}$$

where the index $j$ denotes transverse directions. Since we now have transverse oscillators only, the Hilbert space will be made up of states of the type

$$\text{NS)} \quad \left(\alpha_{-n_1}^{i_1}\cdots\alpha_{-n_k}^{i_k}\right)\left(\psi_{-r_1}^{j_1}\cdots\psi_{-r_q}^{j_q}\right)|0,P\rangle_{\text{NS}}, \tag{6.109a}$$

$$\text{R)} \quad \left(\alpha_{-n_1}^{i_1}\cdots\alpha_{-n_k}^{i_k}\right)\left(\psi_{-r_1}^{j_1}\cdots\psi_{-r_q}^{j_q}\right)|\overset{(\cdot)}{\alpha},P\rangle_{\text{R}}, \tag{6.109b}$$

where, as said above, $|\alpha\rangle$ is the $8_C$ spinorial representation of $SO(8)$, while $|\dot\alpha\rangle$ is the $8_S$ spinorial representation.

If we now implement the lightcone $L_0$ constraint, we have that

$$\text{NS)} \quad L_0^{(\text{lc})} - a_{\text{NS}}^{(\text{lc})} = 0, \tag{6.110a}$$

$$\text{R)} \quad L_0^{(\text{lc})} - a_{\text{R}}^{(\text{lc})} = 0. \tag{6.110b}$$

### 6.2.4.2 Lightcone NS spectrum

Let us now analyze the lightcone NS spectrum up to the massless level.

■ **Level $N_{\text{NS}}^{(\text{lc})} = 0$**
At the level $N_{\text{NS}}^{(\text{lc})} = 0$ we find the tachyon

$$t(P)|0,P\rangle_{\text{NS}}, \tag{6.111}$$

having

$$m^2 = -a_{\text{NS}}^{(\text{lc})}\frac{4}{\alpha'}. \tag{6.112}$$

■ **Level $N_{NS}^{(lc)} = \frac{1}{2}$**

At the level $N_{NS}^{(lc)} = 1/2$ the generic state is

$$\xi_i(P)\psi^i_{-\frac{1}{2}}|0,P\rangle_{NS}\,,\tag{6.113}$$

where, again, the index $i$ denotes transverse directions only. The mass-shell is

$$\frac{\alpha' m^2}{4} = \frac{1}{2} - a_{NS}^{(lc)}\,.\tag{6.114}$$

Such a state is a massless vector in the $8_V$ representation of the Lorentz group $SO(8)$, since it has $d-2$ d.o.f. (which are the transverse directions). Therefore it has to be massless, which implies

$$a_{NS}^{(lc)} = \frac{1}{2}\,,\tag{6.115}$$

which is the same result we found before (see eq. (6.82)).

If we try to directly normal order $L_0$ we have that

$$\begin{aligned}
L_0^{(lc)} &= \frac{\alpha' P^2}{4} + N_\perp \\
&= \frac{\alpha' P^2}{4} + \frac{1}{2}\sum_{k\in\mathbb{Z}}\alpha^i_{-k}\alpha_{k,i} + \frac{1}{2}\sum_{r\in\mathbb{Z}+\frac{1}{2}} r\psi^i_{-r}\psi_{r,i} \\
&= \frac{\alpha' P^2}{4} + \sum_{k\geq 1}\left(\alpha^i_{-k}\alpha_{k,i} + \frac{1}{2}k(d-2)\right) + \sum_{r\geq\frac{1}{2}}\left(r\psi^i_{-r}\psi_{r,i} - \frac{1}{2}r(d-2)\right) \\
&= \frac{\alpha' P^2}{4} + \hat{N}_\perp + \underbrace{\frac{1}{2}(d-2)\sum_{k\geq 1}k - \frac{1}{2}(d-2)\sum_{r\geq\frac{1}{2}}r}_{\text{infinite zero point energies}}\,,
\end{aligned}\tag{6.116}$$

where the second step has been performed just like we did in eq. (3.361), with the only difference that in the $\psi$ case we had to take into account its fermionic nature with a proper minus sign. Moreover, we denoted as $\hat{N}_\perp$ the level operator in the transverse directions.

In analogy to the renormalization we performed for the bosonic string, we can write more in general that

$$\sum_{k\geq\nu}k \longrightarrow \sum_{k\geq\nu}ke^{-\epsilon k} = \frac{e^{\epsilon\nu}(\nu + e^{\epsilon(1-\nu)})}{(1-e^\epsilon)^2} \overset{\epsilon\to 0}{=} \frac{1}{\epsilon^2} - \frac{1}{12}(1 - 6\nu + 6\nu^2) + O(\epsilon)\,.\tag{6.117}$$

This infinite sum can be also understood as a non convergent representation of the Hurwitz zeta-function

$$\zeta(s,\nu) = \sum_{n=0}^\infty (n+\nu)^{-s}\,, \qquad \zeta(-1,\nu) = -\frac{1}{12}(1 - 6\nu + 6\nu^2)\,.\tag{6.118}$$

In the $X$-CFT case we have $\nu = 1$, while in the $\psi$-CFT case we have $\nu = 1/2$. If we now renormalize the action inserting a proper counterterm that absorbs the $1/\epsilon^2$ divergence (as we did in (3.364)), we can write the renormalized zero point energies as

$$\sum_{k\geq 1}k = -\frac{1}{12}\,, \qquad \sum_{r\geq\frac{1}{2}}r = \frac{1}{24}\,.\tag{6.119a}$$

Therefore

$$L_0^{(\text{lc})} = \frac{\alpha' P^2}{4} + \hat{N}_\perp - \frac{d-2}{16}, \tag{6.120}$$

and the constraint $L_0^{(\text{lc})} - a_{\text{NS}}^{(\text{lc})} = 0$ imposes

$$a_{\text{NS}}^{(\text{lc})} = \frac{d-2}{16} = \frac{1}{2} \implies d = 10. \tag{6.121}$$

### 6.2.4.3  Lightcone R spectrum

Let us now analyze the lightcone R spectrum up to the massless level.

■ **Level** $N_{\text{R}}^{(\text{lc})} = 0$

At the level $N_{\text{R}}^{(\text{lc})} = 0$ the generic state is the massless $8_C + 8_S$:

$$u_\alpha(P) |\alpha, P\rangle_{\text{R}} + v_{\dot\alpha}(P) |\dot\alpha, P\rangle_{\text{R}} . \tag{6.122}$$

We can write

$$\begin{aligned}
L_0^{(\text{lc})} &= \frac{\alpha' P^2}{4} + \frac{1}{2} \sum_{k \in \mathbb{Z}} \alpha^i_{-k} \alpha_{k,i} + \frac{1}{2} \sum_{n \in \mathbb{Z}} n \psi^i_{-n} \psi_{n,i} \\
&= \frac{\alpha' P^2}{4} + \hat{N}_\perp + \frac{1}{2}(d-2)\!\!\!\bcancel{\sum_{k \geq 1} k} - \frac{1}{2}(d-2)\!\!\!\bcancel{\sum_{n \geq 1} n} .
\end{aligned} \tag{6.123}$$

We notice that zero point energies cancel each other.

The constraint $L_0^{(\text{lc})} - a_{\text{R}}^{(\text{lc})} = 0$ imposes

$$a_{\text{R}}^{(\text{lc})} = 0 . \tag{6.124}$$

Notice that this is a different value from the one we adopted in the closed superstring OCQ (see eq. (6.82)).[35]

### 6.2.5  The tachyon and the problem with modular invariance

Considering what we said so far for the holomorphic closed superstring spectrum, we can apply it to the anti-holomorphic sector too in order to find the full closed-string spectrum. The result is shown in table 6.2.

We notice that the annoying tachyon is still there. But this is not the real problem. It is not obvious that the theory is consistent at one-loop. In particular the one-loop partition function must be modular invariant, and this is a constraint on the states inside the spectrum. The full one-loop path integral in the matter-ghost sector is subject to an analogous lightcone reduction as in the bosonic string and, in full analogy to the bosonic string, it is given by

$$Z_{\text{1-loop}} = \int_{\mathcal{F}} \frac{d\tau d\bar\tau}{(\text{Im}(\tau))^2} \, \text{Tr}_\perp \left[ (-1)^{\mathbb{F}} q^{L_0 - \frac{c}{24}} \bar{q}^{\bar{L}_0 - \frac{c}{24}} \right], \tag{6.125}$$

where, since now we have both spacetime bosons and fermions circulating in the loop, we put a $(-1)^{\mathbb{F}}$ where $\mathbb{F}$ is the *space-time* fermion number (not to be confused with the *world-sheet* fermion number, which we will introduce later), to account for the standard minus sign for spacetime fermions circulating in the loop. The measure $d^2\tau/(\text{Im}\tau)^2$ is already modular

---

[35]The reason is that the lightcone value is computing the full conformal weight of the R vacuum state in the matter-ghost theory, $\sim c \, e^{-\varphi/2} S_\alpha |0\rangle_{SL(2,\mathbb{C})}$.

Table 6.2: The full closed superstring spectrum up to the massless level.

| | $NS - \overline{NS}$ | $R - \overline{NS}$ | $NS - \overline{R}$ | $R - \overline{R}$ |
|---|---|---|---|---|
| $L_0 - a = 0$ <br> $\bar{L}_0 - a = 0$ | $m^2 = \frac{4}{\alpha'}\left(N_{NS} - \frac{1}{2}\right)$ <br><br> $N_{NS} = \bar{N}_{NS}$ | $m^2 = \frac{4}{\alpha'}N_R$ <br><br> $N_R = \bar{N}_{NS} - \frac{1}{2}$ | $m^2 = \frac{4}{\alpha'}\bar{N}_R$ <br><br> $\bar{N}_R = N_{NS} - \frac{1}{2}$ | $m^2 = \frac{4}{\alpha'}\bar{N}_R$ <br><br> $N_R = \bar{N}_R$ |
| $N = 0$ | tachyon: <br> $t(P)\|0,P\rangle$ <br> $m^2 = -\frac{2}{\alpha'}$ | | | |
| $N_{NS} = \frac{1}{2}$ <br><br> $N_R = 0$ | massless boson: <br> $\underbrace{8_V}_{\text{bos}} \otimes \underbrace{8_V}_{\text{bos}}$ <br> $m^2 = 0$ | massless fermion: <br> $\underbrace{(8_C \oplus 8_S)}_{\text{ferm}} \otimes \underbrace{8_V}_{\text{bos}}$ <br> $m^2 = 0$ | massless fermion: <br> $\underbrace{8_V}_{\text{bos}} \otimes \underbrace{(8_C \oplus 8_S)}_{\text{ferm}}$ <br> $m^2 = 0$ | massless boson: <br> $\underbrace{(8_C \oplus 8_S)}_{\text{ferm}} \otimes \underbrace{(8_C \oplus 8_S)}_{\text{ferm}}$ <br> $m^2 = 0$ |

invariant and contains as usual the volume of the CKG and the momentum integral in the lightcone directions $d^2 p^{\pm}$. The remaining trace is over the transverse $c = 8 + 4$ $(X, \psi)$ SCFT.

$$L_0 - \frac{c}{24} = \left(L_0^X - \frac{8}{24}\right) + \left(L_0^\psi - \frac{4}{24}\right). \tag{6.126}$$

Specializing on the NS or R sector for the 8 transverse $\psi$'s

$$\text{NS)} \quad L_{0\ NS}^\psi = N_{NS}, \tag{6.127a}$$

$$\text{R)} \quad L_{0\ R}^\psi = N_R + \frac{8}{16} = N_R + \frac{1}{2}, \tag{6.127b}$$

where we have considered that in the $R$ sector all of the states are build on the $R$ vacua $|a\rangle_R$ which obey

$$L_0 |a\rangle_R = \frac{1}{2} |a\rangle_R. \tag{6.128}$$

Let's then define

$$
\begin{aligned}
f(\tau, \bar{\tau}) &= \text{Tr}_\perp\left[(-1)^{\mathbb{F}} q^{L_0 - \frac{c}{24}} \bar{q}^{\bar{L}_0 - \frac{c}{24}}\right] \\
&= \text{Tr}_X\left[q^{L_0 - \frac{1}{3}} \bar{q}^{\bar{L}_0 - \frac{1}{3}}\right] \left(\text{Tr}_{NS}\left[q^{N - \frac{1}{6}}\right] - \text{Tr}_R\left[q^{N + \frac{1}{3}}\right]\right) \left(\text{Tr}_{\overline{NS}}\left[\bar{q}^{\bar{N} - \frac{1}{6}}\right] - \text{Tr}_{\overline{R}}\left[\bar{q}^{\bar{N} + \frac{1}{3}}\right]\right) \\
&= \left(\frac{1}{\sqrt{\text{Im}\tau}\,\eta(\tau)\,\eta(\bar{\tau})}\right)^8 \left(\text{Tr}_{NS}\left[q^{N - \frac{1}{6}}\right] - \text{Tr}_R\left[q^{N + \frac{1}{3}}\right]\right) \left(\text{Tr}_{\overline{NS}}\left[\bar{q}^{\bar{N} - \frac{1}{6}}\right] - \text{Tr}_{\overline{R}}\left[\bar{q}^{\bar{N} + \frac{1}{3}}\right]\right),
\end{aligned} \tag{6.129}
$$

where we have calculated the trace over 8 free bosons $X^i$, recalling the Dedekind $\eta$ function

$$\eta(\tau) = q^{\frac{1}{24}} \prod_{n=1}^{\infty}(1 - q^n), \qquad q = e^{2\pi i \tau}. \tag{6.130}$$

SciPost Phys. Lect. Notes 90 (2025)

It is easy to compute the fermionic traces. We start with

$$\text{Tr}_{\text{NS}}\left[q^{N-\frac{1}{6}}\right] = q^{-\frac{1}{6}}\left(\prod_{r\geq\frac{1}{2}}(1+q^r)\right)^8 = \left(\frac{\prod_{n=1}^{\infty}(1-q^n)\left(1+q^{n-\frac{1}{2}}\right)^2}{\eta(\tau)}\right)^4$$

$$\equiv \frac{\theta_3(\tau)^4}{\eta(\tau)^4}\,, \tag{6.131}$$

where we have defined the first of the $\theta$-functions, $\theta_3(\tau)$. Continuing with the $R$ sector we have

$$\text{Tr}_{\text{R}}\left[q^{N+\frac{1}{3}}\right] = q^{-\frac{1}{6}+\frac{1}{2}}\left(\prod_{n\geq 0}(1+q^n)\right)^8 = \left(\frac{q^{\frac{1}{8}}\prod_{n=1}^{\infty}(1-q^n)(1+q^n)^2(1+q^0)^2}{\eta(\tau)}\right)^4$$

$$\equiv \frac{\theta_2(\tau)^4}{\eta(\tau)^4}\,, \tag{6.132}$$

where $\theta_2(\tau)$ has been defined.

To understand the behaviour of these two traces under modular transformations we need to complete the picture with the remaining two $\theta$ functions $\theta_4$ and $\theta_1$. These two rely on the definition of the *worldsheet* fermion number $F$, which is defined as follows

$$\boxed{\begin{aligned} &\text{NS)}\quad (-1)^F = -e^{i\pi\sum_{r\geq\frac{1}{2}}\psi^i_{-r}\psi^i_r}\,, &&(6.133\text{a})\\ &\text{R)}\quad (-1)^F = e^{i\pi\sum_{n\geq 0}\psi^i_{-n}\psi^i_n} \to \Gamma_9\, e^{i\pi\sum_{n\geq 1}\psi^i_{-n}\psi^i_n}\,. &&(6.133\text{b}) \end{aligned}}$$

Few explanations: the reason for the minus sign in the NS sector is that $|0\rangle_{NS}$ secretly contains a fermionic insertion $\delta(\gamma) = e^{-\phi}$ from the ghost $(\beta,\gamma)$ system, over the bosonic $SL(2,\mathbb{C})$ vacuum $|0\rangle_{NS} = e^{-\phi}(0)|0\rangle_{SL(2,\mathbb{C})}$.[36] In the Ramond sector instead we have isolated the zero modes and we have recognized that they are fully captured by the chirality matrix $\Gamma_9$: this is clear since every zero-mode fermionic oscillator is a gamma matrix and we can span the whole zero-mode sector by acting on the conventional vacuum $\left|+\frac{1}{2},+\frac{1}{2},+\frac{1}{2},+\frac{1}{2}\right\rangle$ with all possible gamma matrices, each time changing the worldsheet fermion number by one. Let's see some relevant example, starting from the holomorphic NS vacuum

$$(-1)^F|0,P\rangle_{\text{NS}} = -|0,P\rangle_{\text{NS}}\,. \tag{6.134}$$

So, as anticipated, the (lightcone) NS vacuum is considered to be fermionic. On the first excited state we have

$$(-1)^F\psi^i_{-\frac{1}{2}}|0,P\rangle_{\text{NS}} = +\psi^i_{-\frac{1}{2}}|0,P\rangle_{\text{NS}}\,. \tag{6.135}$$

On the other hand, in the R sector we have that

$$(-1)^F|\alpha,P\rangle_{\text{R}} = +|\alpha,P\rangle_{\text{R}}\,, \tag{6.136a}$$

$$(-1)^F|\dot\alpha,P\rangle_{\text{R}} = -|\dot\alpha,P\rangle_{\text{R}}\,. \tag{6.136b}$$

With this understanding, we can now compute

$$\text{Tr}_{\text{NS}}\left[(-1)^F q^{N-\frac{1}{6}}\right] = -q^{-\frac{1}{6}}\left(\prod_{r\geq\frac{1}{2}}(1-q^r)\right)^8 = -\left(\frac{\prod_{n=1}^{\infty}(1-q^n)\left(1-q^{n-\frac{1}{2}}\right)^2}{\eta(\tau)}\right)^4$$

$$\equiv -\frac{\theta_4(\tau)^4}{\eta(\tau)^4}\,, \tag{6.137}$$

---

[36]As you can see, some residual super-ghost contamination remains even in the lightcone, see section 6.5.1 for details.

and

$$\mathrm{Tr_R}\left[(-1)^F q^{N+\frac{1}{3}}\right] = q^{-\frac{1}{6}+\frac{1}{2}}\left(\prod_{n\geq 0}(1-q^n)\right)^8 = \left(\frac{q^{\frac{1}{8}}\prod_{n=1}^{\infty}(1-q^n)(1+q^n)^2(1-q^0)^2}{\eta(\tau)}\right)^4$$

$$\equiv \frac{\theta_1(\tau)^4}{\eta(\tau)^4} \equiv 0. \tag{6.138}$$

This last trace[37] is indeed identically zero because of the cancellation $(1-q^0)^2 = (1-1)^2$ in the zero mode sector:

$$\theta_1(\tau) \equiv 0. \tag{6.139}$$

Now we are ready to discuss the modular invariance. The modular properties of the $\theta$ functions (together with the $\eta$) are given by

$$\theta_2(\tau+1) = e^{i\frac{\pi}{4}}\theta_2(\tau), \tag{6.140a}$$

$$\theta_3(\tau+1) = \theta_4(\tau), \tag{6.140b}$$

$$\theta_4(\tau+1) = \theta_3(\tau), \tag{6.140c}$$

$$\eta(\tau+1) = e^{i\frac{\pi}{12}}\eta(\tau), \tag{6.140d}$$

and

$$\theta_2\left(-\frac{1}{\tau}\right) = (-i\tau)^{\frac{1}{2}}\theta_4(\tau), \tag{6.141a}$$

$$\theta_3\left(-\frac{1}{\tau}\right) = (-i\tau)^{\frac{1}{2}}\theta_3(\tau), \tag{6.141b}$$

$$\theta_4\left(-\frac{1}{\tau}\right) = (-i\tau)^{\frac{1}{2}}\theta_2(\tau), \tag{6.141c}$$

$$\eta\left(-\frac{1}{\tau}\right) = (-i\tau)^{\frac{1}{2}}\eta(\tau). \tag{6.141d}$$

Using these properties it is immediate to see that the full R-NS partition function of the $c=4$ $\psi$ sector

$$\mathrm{Tr}_\psi\left[(-1)^{\mathbb{F}} q^{L_0-\frac{c}{24}}\bar{q}^{\bar{L}_0-\frac{c}{24}}\right] = \left|\frac{\theta_3(\tau)^4-\theta_2(\tau)^4}{\eta(\tau)^4}\right|^2, \tag{6.142}$$

*is not modular invariant!* Luckily this is not a dead-end and there is a way-out, the *GSO*-projection (Gliozzi, Sherk, Olive).

## 6.3 GSO projection

Given the operator $(-1)^F$ in the NS and R sector, we can build the projectors

$$\mathcal{P}_{\pm}^{(\mathrm{R,NS})} = \frac{1}{2}\left(1\pm(-1)^{F_{(\mathrm{R,NS})}}\right). \tag{6.143}$$

Notice that in the R sector the two $\pm$ choices simply change the selected chirality of the space-time fermions at every level. Using these projectors in the holomorphic and anti-holomorphic sector of the $\psi$ CFT it has been understood (and you can easily check) that there are four possible truncations of the spectrum which are modular invariant.

---

[37]The four traces we have computed correspond to the possible periodicities that we can have for a fermion around the euclidean time circle and the space circle (which gives the already discussed R, NS degeneracy). Assigning a given periodicity around all possible independent cycles of a Riemann surface means to give a *spin structure*. On the torus we have a total of four different spin structures and each of them is giving a trace.

- Type IIA or Type IIB.

  These are two projections that are done separately on each holomorphic sector

  $$\mathcal{P}_{IIA} \equiv \left(\mathcal{P}_+^{(NS)} + \mathcal{P}_+^{(R)}\right)\overline{\left(\mathcal{P}_+^{(NS)} + \mathcal{P}_-^{(R)}\right)}, \tag{6.144}$$

  $$\mathcal{P}_{IIB} \equiv \left(\mathcal{P}_+^{(NS)} + \mathcal{P}_+^{(R)}\right)\overline{\left(\mathcal{P}_+^{(NS)} + \mathcal{P}_+^{(R)}\right)}. \tag{6.145}$$

  They give rise to the same partition function[38]

  $$\mathrm{Tr}_\psi\left[\mathcal{P}_{IIA}(-1)^{\mathbb{F}}q^{L_0-\frac{c}{24}}\bar{q}^{\bar{L}_0-\frac{c}{24}}\right] = \mathrm{Tr}_\psi\left[\mathcal{P}_{IIB}(-1)^{\mathbb{F}}q^{L_0-\frac{c}{24}}\bar{q}^{\bar{L}_0-\frac{c}{24}}\right]$$
  $$= \left|\frac{\theta_3(\tau)^4 - \theta_4(\tau)^4 - \theta_2(\tau)^4}{\eta(\tau)^4}\right|^2, \tag{6.146}$$

  which is easily shown to be modular invariant. There is one important fact that we have to know about this partition function: it is *identically vanishing!* This is because of Jacobi's *aequatio identica satis abstrusa* (friendly referred to as Jacobi's abstruse identity):

  $$\theta_3(\tau)^4 - \theta_4(\tau)^4 - \theta_2(\tau)^4 = 0. \tag{6.147}$$

  The physical meaning of this is easy to grasp: we are considering a partition function that computes the number of bosonic states minus the fermionic states. Since we find zero it means that we have a perfect pairing of bosonic and fermionic degrees of freedom. This is the sign of *spacetime Supersymmetry*. Another related thing, as we will shortly see, is that the NS-NS tachyon state has been projected out. These are thus the first examples of stable string theory vacua. Collecting also the $X$ sector (which is already modular invariant by itself, as we have seen in the bosonic string) we finally have

  $$Z_{1\text{-loop}}^{IIA,B} = \int_{\mathcal{F}} \frac{d\tau d\bar{\tau}}{(\mathrm{Im}(\tau))^2} \mathrm{Tr}_\perp\left[\mathcal{P}_{IIA,B}(-1)^{\mathbb{F}}q^{L_0-\frac{1}{2}}\bar{q}^{\bar{L}_0-\frac{1}{2}}\right] \tag{6.148}$$
  $$= \int_{\mathcal{F}} \frac{d\tau d\bar{\tau}}{(\mathrm{Im}(\tau))^2}\left(\frac{1}{\sqrt{\mathrm{Im}\tau}\,\eta(\tau)\,\eta(\bar{\tau})}\right)^8 \left|\frac{\theta_3(\tau)^4 - \theta_4(\tau)^4 - \theta_2(\tau)^4}{\eta(\tau)^4}\right|^2 = 0.$$

  Since this loop amplitude is computing the space-time vacuum energy due to the formation of bubbles, this is saying that there is no space-time cosmological constant that is generated by vacuum fluctuations. This is again due to the precise microscopic cancellations implied by space-time supersymmetry.

- Type 0A or Type 0B.

  The other two possibilities are given by the following projections

  $$\mathcal{P}_{0A} \equiv \mathcal{P}_+^{(NS)}\overline{\mathcal{P}_+^{(NS)}} + \mathcal{P}_-^{(NS)}\overline{\mathcal{P}_-^{(NS)}} + \mathcal{P}_+^{(R)}\overline{\mathcal{P}_-^{(R)}} + \mathcal{P}_-^{(R)}\overline{\mathcal{P}_+^{(R)}}, \tag{6.149}$$

  $$\mathcal{P}_{0B} \equiv \mathcal{P}_+^{(NS)}\overline{\mathcal{P}_+^{(NS)}} + \mathcal{P}_-^{(NS)}\overline{\mathcal{P}_-^{(NS)}} + \mathcal{P}_+^{(R)}\overline{\mathcal{P}_+^{(R)}} + \mathcal{P}_-^{(R)}\overline{\mathcal{P}_-^{(R)}}. \tag{6.150}$$

One thing is immediately noticed: in this truncation there are only space-time bosons and fermions have been completely projected out. Second, the tachyon is still there since it is selected by $\mathcal{P}_-^{(NS)}\overline{\mathcal{P}_-^{(NS)}}$. The two partition functions are again identical (because of $\theta_1 = 0$)

$$\mathrm{Tr}_\psi\left[\mathcal{P}_{0A}q^{L_0-\frac{c}{24}}\bar{q}^{\bar{L}_0-\frac{c}{24}}\right] = \mathrm{Tr}_\psi\left[\mathcal{P}_{0B}q^{L_0-\frac{c}{24}}\bar{q}^{\bar{L}_0-\frac{c}{24}}\right]$$
$$= \frac{|\theta_3(\tau)|^8 + |\theta_4(\tau)|^8 + |\theta_2(\tau)|^8}{|\eta(\tau)|^8}, \tag{6.151}$$

---

[38]This is because $\theta_1(\tau) \equiv 0$ and it is an accident of the fact that we are computing a vacuum amplitude with no vertex-operator insertions. The generic Type IIA-B loop amplitudes are of course different as they describe interactions of two different set of states, as we will see shortly.

and the full loop vacuum amplitude is

$$Z_{1\text{-loop}}^{0A,B} = \int_{\mathcal{F}} \frac{d\tau d\bar{\tau}}{(\text{Im}(\tau))^2} \, \text{Tr}_{\perp} \left[ \mathcal{P}_{0A,B} \, q^{L_0 - \frac{1}{2}} \, \bar{q}^{\bar{L}_0 - \frac{1}{2}} \right],$$

$$= \int_{\mathcal{F}} \frac{d\tau d\bar{\tau}}{(\text{Im}(\tau))^2} \left( \frac{1}{\sqrt{\text{Im}\tau} \, \eta(\tau) \, \eta(\bar{\tau})} \right)^8 \frac{|\theta_3(\tau)|^8 + |\theta_4(\tau)|^8 + |\theta_2(\tau)|^8}{|\eta(\tau)|^8}.$$

(6.152)

Both Type 0A and 0B theories are qualitatively similar to the bosonic string: no fermions and an unstable vacuum signaled by the closed string tachyon. In the sequel we will not consider them anymore.

## 6.4 Type II superstring

### 6.4.1 Closed string

Let us keep studying the Type II closed superstring. Given what we said about the GSO projection, we now write $\text{GSO}^{\pm}$ to identify the GSO projection in the holomorphic sector and $\overline{\text{GSO}}^{\pm}$ to identify the GSO projection in the anti-holomorphic sector. Hence we have that the choice of the Ramond GSO projection generates two inequivalent theories:

- type IIA, which corresponds to the choice $\left( \text{GSO}_{\text{R}}^{+}, \overline{\text{GSO}}_{\text{R}}^{-} \right)$,

- type IIB, which corresponds to the choice $\left( \text{GSO}_{\text{R}}^{+}, \overline{\text{GSO}}_{\text{R}}^{+} \right)$.

Obviously, they both are $\left( \text{GSO}_{\text{NS}}^{+}, \overline{\text{GSO}}_{\text{NS}}^{+} \right)$ in the NS sector, so that the $\text{NS} - \overline{\text{NS}}$ tachyon is removed. We are now going to explicitly see that these two theories have an equal number of bosonic and fermionic d.o.f. at every mass level, which is an explicit signal of spacetime supersymmetry.

#### 6.4.1.1 Closed type IIA

If we now analyze the closed type IIA spectrum up to the massless level in the lightcone quantization, we find what follows.

• **NS $-\overline{\text{NS}}$ sector**
The generic polarization of this sector is a Lorentz representation with two $8_V$ indices which can be split up into its symmetric part (s), anti-symmetric part (a) and trace:

$$t^{ij} = \delta^{ij}\Phi + a^{[ij]} + s^{(ij)}.$$

(6.153)

This can be written as

$$8_V \otimes 8_V = 1 \oplus 28_a \oplus 35_s,$$

(6.154)

where

· the singlet (1) is the dilaton (spin 0),

· the anti-symmetric part ($28_a$) is the Kalb-Ramond field (spin 1),

· the symmetric part ($35_s$) is the graviton (spin 2).

- **NS $-\bar{\mathrm{R}}$ sector**

The generic polarization of this sector has a vector index $\mu$ and a spinorial index $\dot{\alpha}$, namely $\chi_{\dot{\alpha}}^i$. This can be written as

$$8_V \otimes 8_S = 8_C \oplus 56_S\,, \tag{6.155}$$

where

· the $8_C$ is the dilatino (spin 1/2, left in spacetime), which is the $\Gamma$-trace of $\chi_{\dot{\alpha}}^i$:

$$\eta_\beta = \Gamma_{\beta\dot{\alpha}}^i \chi_{\dot{\alpha}}^i\,, \tag{6.156}$$

· the $56_S$ is the gravitino (spin 3/2, right in spacetime):

$$\psi_{\dot{\beta}}^i = \chi_{\dot{\beta}}^i - \frac{1}{D-2}\Gamma_{\dot{\beta}\alpha}^i \eta_\alpha\,. \tag{6.157}$$

- **R $-\overline{\mathrm{NS}}$ sector**

The generic polarization of this sector has a vector index $\mu$ and a spinorial index $\alpha$, namely $\chi_\alpha^i$. This means

$$8_C \otimes 8_V = 8_S \oplus 56_C\,, \tag{6.158}$$

where

· the $8_S$ is the dilatino (spin 1/2, right in spacetime), which is the $\Gamma$-trace of $\chi_\alpha^i$:

$$\eta_{\dot{\beta}} = \Gamma_{\dot{\beta}\alpha}^i \chi_\alpha^i\,, \tag{6.159}$$

· the $56_C$ is the gravitino (spin 3/2, left in spacetime):

$$\psi_{\dot{\beta}}^i = \chi_{\dot{\beta}}^i - \frac{1}{D-2}\Gamma_{\dot{\beta}\dot{\alpha}}^i \eta_{\dot{\alpha}}\,. \tag{6.160}$$

- **R $-\bar{\mathrm{R}}$ sector**

The generic polarization of this sector has two spinorial indices $\alpha, \dot{\alpha}$, namely $H_{\alpha\dot{\alpha}}$. It can be expanded in $\Gamma$ matrices as follows:

$$H_{\alpha\dot{\alpha}} = C_i \Gamma_{\alpha\dot{\alpha}}^i + C_{ijk}\Gamma_{\alpha\dot{\alpha}}^{ijk}\,, \tag{6.161}$$

where we used the definition

$$\Gamma^{i_1\dots i_n} = \frac{1}{n!}\Gamma^{[i_1}\cdots\Gamma^{i_n]}\,. \tag{6.162}$$

The reason why, inside the decomposition (6.161), we did not go further in the differential forms order will be clarified in 6.4.1.3. In representation notation, this expansion can be written as

$$8_C \otimes 8_S = 8_V \oplus 56_a\,, \tag{6.163}$$

where

· the $8_V$ is the massless vector $C_\mu$ (spin 1), which is a 1-form whose associated field strength is $F^{(2)} = dC^{(1)}$, namely

$$F_{\mu\nu} = \partial_{[\mu}C_{\nu]}\,, \tag{6.164}$$

· the anti-symmetric $56_a$ is the 3-form $C_{\mu\nu\rho}$ whose associated field strength is $F^{(4)} = dC^{(3)}$, namely

$$F_{\mu\nu\rho\sigma} = \partial_{[\mu}C_{\nu\rho\sigma]}\,. \tag{6.165}$$

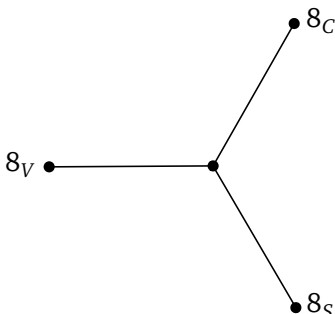

Figure 6.2: The $SO(8)$ algebra Dynkin diagram.

Table 6.3: The massless spectrum of the type IIA closed superstring.

| Closed Type IIA | | | | | |
|---|---|---|---|---|---|
| **Bosons** | | | **Fermions** | | |
| $\Phi$ | dilaton | spin 0 | $\eta_\alpha$ | dilatino (left) | spin 1/2 |
| $B_{ij}$ | Kalb-Ramond | spin 1 | $\eta_{\dot\alpha}$ | dilatino (right) | spin 1/2 |
| $G_{ij}$ | graviton | spin 2 | $\psi^i_\beta$ | gravitino (left) | spin 3/2 |
| $C_i$ | 1-form | | $\psi^i_{\dot\beta}$ | gravitino (right) | spin 3/2 |
| $C_{ijk}$ | 3-form | | | | |

We used the notation $C^{(p)} = C_{i_1 \dots i_p} dx^{i_1} \wedge \cdots \wedge dx^{i_p}$. These differential forms are called *Ramond-Ramond forms* and we will investigate them more in 6.4.1.3.

Everything we said so far about $SO(8)$ representations is deeply related to its *triality* property.[39]

The massless content of the type IIA closed superstring is summarized in table 6.3.

It is the same spectrum we find in type IIA supergravity (SUGRA), a theory invariant under local supersimmetry which leads to general coordinate invariance (since, as we saw in eq. (6.34), (SUSY)$^2$ = translation).

Its action is

$$S_{\text{IIA}} = \frac{1}{2K_{10}^2} \int d^{10}x \sqrt{-G} \left\{ e^{-2\Phi} \left( R + 4(\partial\Phi)^2 - \frac{1}{2}|H_{(3)}|^2 \right) - \frac{1}{2}|F_{(2)}|^2 - \frac{1}{2}|F_{(4)}|^2 \right\}$$

$$+ \frac{1}{4K_{10}^2} \int B_{(2)} \wedge dC_{(3)} \wedge dC_{(3)}$$

$$+ \text{(fermions couplings)}$$

$$+ (\alpha' \text{ corrections}), \tag{6.167}$$

---

[39]The $SO(8)$ group has a triality property, which means that its vectorial representation and its two spinorial representations, having the same dimension, are isomorphic. We can write

$$8_i \otimes 8_i = 1 \oplus 28 \oplus 35, \quad \text{with } i = V, C, S, \tag{6.166a}$$

$$8_i \otimes 8_j = 8_k \oplus 56, \quad \text{with } i \neq j \neq k. \tag{6.166b}$$

Such a property is evocative of the explicit $\mathbb{Z}_3$-symmetry of the Dynkin diagram of the $SO(8)$ algebra (see fig. 6.2).

Table 6.4: The massless spectrum of the type IIB closed superstring.

| Closed Type IIB | | | | | | |
|---|---|---|---|---|---|---|
| Bosons | | | Fermions | | | |
| $\Phi$ | dilaton | spin 0 | $\eta_{\dot{\alpha}}$ | dilatino (right) | spin 1/2 | |
| $B_{ij}$ | Kalb-Ramond | spin 1 | $\eta'_{\dot{\alpha}}$ | dilatino (right) | spin 1/2 | |
| $G_{ij}$ | graviton | spin 2 | $\psi^i_\beta$ | gravitino (left) | spin 3/2 | |
| $C$ | 0-form | spin 0 | $\psi'^i_\beta$ | gravitino (left) | spin 3/2 | |
| $C_{ij}$ | 2-form | | | | | |
| $C_{ijkl}$ | 4-form | | | | | |

where

- $K_{10}$ is the Newton constant in 10 dimensions, which is related to the basic string parameters as $K_{10} = 8\pi^{7/2}\alpha'^2 g_s$,

- $\int d^{10}x\,\sqrt{-G}\,e^{-2\phi}R$ is the Einstein-Hilbert action,

- $4(\partial\Phi)^2$ is the kinetic term for the dilaton,

- $H_{(3)} = dB_{(2)}$ is the kinetic term of the Kalb-Ramond field,

- $F_{(2)} = dC_{(1)}$ and $F_{(4)} = dC_{(3)} - dB_{(2)} \wedge C_{(1)}$ are the kinetic terms for the RR fields,

- $\int B_{(2)} \wedge dC_{(3)} \wedge dC_{(3)}$ is a topological term, independent of the metric.

### 6.4.1.2  Closed type IIB

If we now look at the closed type IIB spectrum up to the massless level in the lightcone quantization, we realize that, mutatis mutandis, is quite similar to the type IIA spectrum. It is summarized in table 6.4.

It is the same spectrum we find in type IIB supergravity, whose action is

$$
S_{\text{IIB}} = \frac{1}{2K_{10}^2}\int d^{10}x\,\sqrt{-G}\left\{e^{-2\Phi}\left(R + 4(\partial\Phi)^2 - \frac{1}{2}|H_{(3)}|^2\right)\right.
$$
$$
\left. -\frac{1}{2}|F_{(1)}|^2 - \frac{1}{2}|F_{(3)}|^2 - \frac{1}{2}|F_{(5)}|^2\right\}
$$
$$
+\frac{1}{4K_{10}^2}\int C_{(4)} \wedge H_{(3)} \wedge F_{(3)}
$$
$$
+ (\text{fermions couplings})
$$
$$
+ (\alpha' \text{ corrections}), \tag{6.168}
$$

where everything has an analogous meaning to what we previously said, and where

$$
F_{(1)} = dC, \tag{6.169a}
$$
$$
F_{(3)} = dC_{(2)} - C\,dB_{(2)}, \tag{6.169b}
$$
$$
F_{(5)} = dC_{(4)} - \frac{1}{2}dC_{(2)} \wedge B_{(2)} + \frac{1}{2}B_{(2)} \wedge dC_{(2)}. \tag{6.169c}
$$

### 6.4.1.3 Ramond-Ramond potentials

Let us now consider the type IIA case. In the $R\bar{R}$ sector the generic state can be expanded in differential forms as

$$H_{\alpha\dot{\alpha}} = C_i \Gamma^i_{\alpha\dot{\alpha}} + C_{i_1\dots i_3} \Gamma^{i_1\dots i_3}_{\alpha\dot{\alpha}} + C_{i_1\dots i_5} \Gamma^{i_1\dots i_5}_{\alpha\dot{\alpha}} + C_{i_1\dots i_7} \Gamma^{i_1\dots i_7}_{\alpha\dot{\alpha}}\,. \tag{6.170}$$

However, thanks to the $\Gamma$ duality property

$$\Gamma^{i_1\dots i_n} \propto \varepsilon^{i_1\dots i_n j_1\dots j_{8-n}} \Gamma_{j_1\dots j_{8-n}}\,, \tag{6.171}$$

these differential forms are not independent with respect to each other. Indeed we have that

$$C_{i_1\dots i_7} \propto \varepsilon_{i_1\dots i_7}{}^j C_j \implies C_{(7)} = \star_8 C_{(1)}\,, \tag{6.172a}$$

$$C_{i_1\dots i_3} \propto \varepsilon_{i_1\dots i_3}{}^{j_1\dots j_5} C_{j_1\dots j_5} \implies C_{(3)} = \star_8 C_{(5)}\,, \tag{6.172b}$$

where $\star_8$ is the Hodge duality in 8 dimensions. We call *Ramond-Ramond potentials* the two independents forms $C_{(1)}$ and $C_{(3)}$. The other two, $C_{(5)}$ and $C_{(7)}$ are not independent and therefore they are not needed in the expansion of the bispinor $H_{\alpha\dot{\alpha}}$. This is the reason why we did not write them in eq. (6.161).

Let's now extend this reasoning from 8 to 10 dimensions. Let's start recalling the form of the Ramond vacua as a spinor-values spin-field acting on the conformal vacuum

$$R) \quad S_\alpha(0)|0\rangle\,, \quad \text{with } \alpha \in 16_C\,, \tag{6.173a}$$

$$\bar{R}) \quad S_{\dot{\alpha}}(0)|0\rangle\,, \quad \text{with } \dot{\alpha} \in 16_S\,. \tag{6.173b}$$

The gamma matrices are now $32\times32$. Moreover, we define $\mathcal{C}$ as the charge conjugation matrix, which realizes a similarity transformation in the Clifford algebra:

$$\mathcal{C}^{-1}(\Gamma^\mu)^T \mathcal{C} = \Gamma^\mu\,. \tag{6.174}$$

The spinors $S_\alpha$ that are involved in the VO defining the Ramond states satisfy the reality condition

$$S_\alpha^\dagger \Gamma^0 = S_\alpha^T \mathcal{C}\,. \tag{6.175}$$

This means that they are Majorana spinors and hence, while performing the transformation $|S_\alpha\rangle \longrightarrow \langle S_\alpha|$, we do not have to worry about taking the hermitian or the BPZ conjugation.

Let us now write the vertex operator

$$F_{\mu_1\dots\mu_p}(P)\Big( (S(z))^T \mathcal{C}\Gamma^{\mu_1\dots\mu_p} \tilde{S}(\bar{z}) \Big)\,. \tag{6.176}$$

If we recall the action of the chirality matrix (in $D = 10$) $\Gamma^{11}$ on the spin fields of Type IIA

$$(\Gamma^{11} S)_\alpha = +S_\alpha\,, \tag{6.177a}$$

$$(\Gamma^{11} \tilde{S})_{\dot{\alpha}} = -\tilde{S}_{\dot{\alpha}}\,, \tag{6.177b}$$

the fermionic bilinear can be rewritten as

$$(S)^T \mathcal{C}\Gamma^{\mu_1\dots\mu_p} \tilde{S} = (S)^T (\Gamma^{11})^T \mathcal{C}\Gamma^{\mu_1\dots\mu_p} \tilde{S}\,. \tag{6.178}$$

Now, using $\{\Gamma^{11}, \Gamma^\mu\} = 0$ and $\mathcal{C}\Gamma^{11} + (\Gamma^{11})^T \mathcal{C} = 0$, we can write

$$
\begin{aligned}
(S_\alpha)^T \mathcal{C}\Gamma^{\mu_1\dots\mu_p} \tilde{S}_{\dot{\beta}} &= (S_\alpha)^T (\Gamma^{11})^T \mathcal{C}\Gamma^{\mu_1\dots\mu_p} \tilde{S}_{\dot{\beta}} \\
&= (-1)^{p+1} (S_\alpha)^T \mathcal{C}\Gamma^{\mu_1\dots\mu_p} \Gamma^{11} \tilde{S}_{\dot{\beta}} \\
&= (-1)^{p+1} (S_\alpha)^T \mathcal{C}\Gamma^{\mu_1\dots\mu_p} \left( -\tilde{S}_{\dot{\beta}} \right) \\
&= (-1)^p (S_\alpha)^T \mathcal{C}\Gamma^{\mu_1\dots\mu_p} \tilde{S}_{\dot{\beta}}\,. \tag{6.179}
\end{aligned}
$$

Therefore $p$ has to be even. This means that in type IIA a $p$-form $F_{\mu_1\ldots\mu_p}$ can be coupled to the two holomorphic and anti-holomorphic spin fields of opposite chirlaity only if $p = 2n$, for $n \in \mathbb{N}$. On the other hand, in lightcone quantization RR potentials had an odd number of vectorial indices. This strongly suggests that $F$ is the field strenght of the $C$ gauge potentials we have enxountered in the lightcone.

To establish this, let us now consider, in the $R\overline{R}$ sector, the state

$$|\text{state}\rangle = F_{\mu_1\ldots\mu_p}\, (S_\alpha)^T\, \mathcal{C}\Gamma^{\mu_1\ldots\mu_p}\tilde{S}_{\hat{\beta}}\, e^{iP\cdot X}(0,0)\,|0\rangle\,. \tag{6.180}$$

The two super-current constraint give two Dirac operators in the two holomorphic sides

$$G_0\,|\text{state}\rangle = 0 \implies P^\mu\psi_{0\mu} = 0\,, \tag{6.181a}$$

$$\overline{G}_0\,|\text{state}\rangle = 0 \implies P^\mu\bar{\psi}_{0\mu} = 0\,, \tag{6.181b}$$

namely

$$\begin{aligned}
G_0\,|\text{state}\rangle &= P_\mu F_{\mu_1\ldots\mu_p}\, (S_\alpha)^T\, (\Gamma^\mu)^T\, \mathcal{C}\Gamma^{\mu_1\ldots\mu_p}\tilde{S}_{\hat{\beta}}(0,0)\,|0\rangle\\
&= P_\mu F_{\mu_1\ldots\mu_p}\, (S_\alpha)^T\, \mathcal{C}\Gamma^\mu\Gamma^{\mu_1\ldots\mu_p}\tilde{S}_{\hat{\beta}}(0,0)\,|0\rangle\\
&= P_\mu F_{\mu_1\ldots\mu_p}\, (S_\alpha)^T\, \mathcal{C}\big(\Gamma_\mu^{\mu_1\ldots\mu_p} + p\,\delta_\mu^{[\mu_1}\Gamma^{\mu_2\ldots\mu_p]}\big)\tilde{S}_{\hat{\beta}}(0,0)\,|0\rangle\\
&= 0\,,
\end{aligned} \tag{6.182}$$

and similarly for $\overline{G}_0$. Therefore the constraints produce

- the Bianchi identities:

$$P_{[\mu}F_{\mu_1\ldots\mu_p]} = 0 \implies dF^{(p)} = 0\,, \tag{6.183}$$

- the generalized propagating Maxwell equations:

$$P_\mu F^\mu{}_{\mu_1\ldots\mu_p} = 0 \implies d\star_{10} F^{(p)} = 0\,. \tag{6.184}$$

Given a field strength $F^{(p)}$, locally there exists a potential $C^{(p-1)}$ such that $F^{(p)} = dC^{(p-1)}$. This is the reason why we will call them "Ramond-Ramond potentials".

In the lightcone we have $C^{(1)}$, $C^{(3)}$ and their Hodge-duals $C^{(7)}$, $C^{(5)}$. In $D = 10$ they become R-R potentials that we still call $C^{(1)}$, $C^{(3)}$ and $C^{(7)}$, $C^{(5)}$, with their associated field strength $F^{(2)}$, $F^{(4)}$ and $F^{(8)}$, $F^{(6)}$. In this 10-dimensional case we use the field strengths, which are gauge invariants. In the lightcone case, on the other hand, we used the potentials $C^{(p-1)}$ since there we had no gauge symmetry. Moreover

$$F^{(6)} = \star_{10}F^{(4)}\,, \tag{6.185a}$$

$$F^{(8)} = \star_{10}F^{(2)}\,, \tag{6.185b}$$

where we used the general definition of the Hodge duality in $d$ dimensions:

$$\big(\star_{10}\, F^{(p)}\big)_{\mu_1\ldots\mu_{d-p}} = \sqrt{-g}\,\varepsilon_{\mu_1\ldots\mu_{d-p}\nu_1\ldots\nu_p}F^{\nu_1\ldots\nu_p}\,. \tag{6.186}$$

This is clearly not a topological operation: it can only be performed on Riemaniann manifolds, namely on metric-equipped manifolds.

The kinetic term for the field strength is given by $F_{\mu_1 \dots \mu_p} F^{\mu_1 \dots \mu_p}$ (just like we used $F_{\mu\nu}F^{\mu\nu}$ in electromagnetism). If we couple it to gravity, we get

$$
\begin{aligned}
S &= \int d^d x \, \sqrt{-g} \, F_{\mu_1 \dots \mu_p} F^{\mu_1 \dots \mu_p} \\
&= \int d^d x \, \sqrt{-g} \, \left| F^{(p)} \right|^2 \\
&= \int_{\mathcal{M}_d} F^{(p)} \wedge \star F^{(p)} \,,
\end{aligned}
\tag{6.187}
$$

where $\mathcal{M}_d$ is a $d$-dimensional manifold.

### 6.4.1.4   Electric-Magnetic duality

In order to understand which is the physical role of Hodge duality, we can look at the electromagnetic tensor in $d = 4$:

$$
F_{\mu\nu} = \begin{pmatrix} 0 & E_1 & E_2 & E_3 \\ -E_1 & 0 & B_3 & -B_2 \\ -E_2 & -B_3 & 0 & B_1 \\ -E_3 & B_2 & -B_1 & 0 \end{pmatrix}.
\tag{6.188}
$$

Its Hodge-dual is

$$
(\star F)_{\mu\nu} = \varepsilon^{\mu\nu\rho\sigma} F_{\rho\sigma} = \widetilde{F}^{\mu\nu} = \begin{pmatrix} 0 & B_1 & B_2 & B_3 \\ -B_1 & 0 & -E_3 & E_2 \\ -B_2 & E_3 & 0 & E_1 \\ -B_3 & -E_2 & E_1 & 0 \end{pmatrix}.
\tag{6.189}
$$

Therefore the action of the Hodge duality is to swap

$$
\vec{E} \longrightarrow \vec{B} \,,
\tag{6.190a}
$$
$$
\vec{B} \longrightarrow -\vec{E} \,.
\tag{6.190b}
$$

Moreover we can write

- Maxwell equations:
$$
\partial_\mu F^{\mu\nu} = 0 \,,
\tag{6.191}
$$

- Bianchi identities:
$$
\partial_\mu \widetilde{F}^{\mu\nu} = 0 \,.
\tag{6.192}
$$

There is an explicit symmetry between these two equations. If we then introduce an electromagnetic source, Maxwell equations become inhomogeneous and the duality is lost unless we assume the existence of a magnetic current:

$$
\partial_\mu F^{\mu\nu} = j^\nu_{\text{electr.}} \,,
\tag{6.193a}
$$
$$
\partial_\mu \widetilde{F}^{\mu\nu} = j^\nu_{\text{magn.}} \,.
\tag{6.193b}
$$

These equations are invariant under

$$
F \longleftrightarrow \widetilde{F} = \star F \,,
\tag{6.194a}
$$
$$
j_{\text{electr.}} \longleftrightarrow j_{\text{magn.}} \,.
\tag{6.194b}
$$

SciPost Phys. Lect. Notes 90 (2025)

Let us now suppose we have a magnetic monopole. We may recall that for a common electric charge $q_e$ the Gauss theorem says that

$$\int_{\mathcal{S}_2} \star F^{(2)} = \int_{\mathcal{S}_2} \vec{E} \cdot d\vec{s} = q_e \,, \tag{6.195}$$

where $\mathcal{S}_2$ is a 2-sphere surrounding $q_e$. Therefore, for a magnetic charge $q_m$ the Gauss theorem says that

$$\int_{\mathcal{S}_2} F^{(2)} = \int_{\mathcal{S}_2} \vec{B} \cdot d\vec{s} = q_m \,. \tag{6.196}$$

Let us then suppose we have an electric charge $q_e$ moving around in the background generated by the magnetic charge $q_m$. The minimal coupling between $q_e$ and the gauge field $A^{(1)}$ (with $dA^{(1)} = F^{(2)}$) is given by

$$\int_{\gamma} A^{(1)} = \int d\tau \, A_\mu \dot{x}^\mu \,, \tag{6.197}$$

where $\tau$ is the parameter that parametrizes $q_e$'s trajectory $\gamma$. Due to Aharonov-Bohm effect, if a $q$-charged particle described by the wave function $\Psi(\vec{x})$ takes a tour along a closed trajectory $\gamma$ embedded in a gauge field $A^{(1)}$, then $\Psi$ gains a phase $e^{iq \int_\gamma A^{(1)}}$. Therefore, if $q_e$ travels in a closed loop nearby $q_m$ (see fig. 6.3), its wave function transforms as

$$\Psi_{q_e}(\vec{x}) \longrightarrow e^{iq_e \int_\gamma A^{(1)}} \Psi_{q_e}(\vec{x}) \,. \tag{6.198}$$

Moreover, referring to fig. 6.3, we have that

$$\gamma = \partial D_1 = -\partial D_2 \,. \tag{6.199}$$

What we just said (together with the assumption that, being far from the magnetic monopole, we can locally write $dA^{(1)} = F^{(2)}$) allows us to compute the following:

$$\int_{\gamma} A^{(1)} = \int_{\partial D_1} A^{(1)} = \int_{D_1} F^{(2)} \,, \tag{6.200a}$$

$$\int_{\gamma} A^{(1)} = \int_{-\partial D_2} A^{(1)} = -\int_{D_2} F^{(2)} \,, \tag{6.200b}$$

and since $D_1 + D_2 = \mathcal{S}_2$, this all implies that

$$-\int_{D_2} F^{(2)} = -\int_{\mathcal{S}_2} F^{(2)} + \int_{D_1} F^{(2)} \,. \tag{6.201}$$

We thus found an ambiguity in the phase. In order to remove this problem we have to impose

$$e^{-iq_e \int_{S^2} F^{(2)}} = e^{-iq_e q_m} = 1 \,, \tag{6.202}$$

where we have used eq. (6.196). Therefore we obtain

$$\boxed{q_e q_m = 2\pi n \,, \quad \text{with } n \in \mathbb{Z} \,,} \tag{6.203}$$

which is called *Dirac quantization condition*. The smallest electric and magnetic charges that satisfy such a condition (i.e. for $n = 1$) are

$$\hat{q}_e \hat{q}_m = 2\pi \,. \tag{6.204}$$

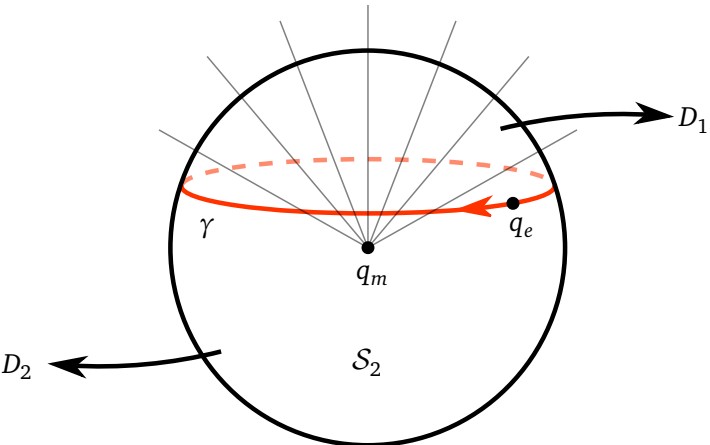

Figure 6.3: The electric charge $q_e$ travelling in a closed loop $\gamma$ around the magnetic charge $q_m$, radiating flux lines.

### 6.4.1.5 Electric and magnetic branes

We saw that a 1-form $A^{(1)}$ naturally couples to a worldline, namely to a particle (of charge $q$), as follows:

$$q \int_\gamma A^{(1)} = q \int d\tau A_\mu \dot{x}^\mu. \tag{6.205}$$

Analogously, a 2-form $B^{(2)}$ naturally couples to a worldsheet, namely to a string. This is the case of the Kalb-Ramond form:

$$\frac{1}{2\pi\alpha'} \int_{\text{WS}} B^{(2)} = \frac{1}{2\pi\alpha'} \int_{\text{WS}} d^2\sigma B_{\mu\nu} \partial X^\mu \bar{\partial} X^\nu. \tag{6.206}$$

This story generalizes for RR fields

- In type IIA theory the R-R potentials are

  - $C^{(1)}$, which couples to an object having a 1-dimensional worldvolume and an electric charge given by the flux of the R-R field, namely to a 0-brane (i.e. a particle),
  - its Hodge-dual $C^{(7)}$, which couples to 6-brane,
  - $C^{(3)}$, which couples to a 2-brane,
  - its Hodge-dual $C^{(5)}$, which couples to 4-brane.

- In type IIB theory the R-R potentials are

  - $C^{(0)}$, which couples to a (-1)-brane,
  - its Hodge-dual $C^{(8)}$, which couples to 7-brane,
  - $C^{(2)}$, which couples to a 1-brane,
  - its Hodge-dual $C^{(6)}$, which couples to 5-brane,
  - $C^{(4)}$, which couples to a 3-brane and which is self-dual.

Polchinski realized that these are precisely the open string $D(p)$-branes! We could then say that the $D(p)$-brane boundary conditions of the $(X; \psi)$-SCFT corresponds to R-R field sources.

This means that, in superstring theory, $D(p)$-branes not only have mass and energy, but also a charge, called *R-R charge*.

A $D(p)$-brane can be described through a boundary state $|D(p)\rangle$ defined as

$$\langle D(p)|\mathcal{V}\rangle = \langle \mathcal{V}(0,0)\rangle_{\text{disk}}^{D(p)\,\text{BC}},\tag{6.207}$$

where $\mathcal{V}$ is a generic matter vertex operator. The boundary state of BC $= a$ defined as

$$\langle B_a|\mathcal{V}\rangle = \;\;\;\text{[diagram]}\;\;\; = \langle \mathcal{V}(0,\bar{0})\rangle_{\text{disk}}^{(a)},\tag{6.208}$$

is an object analogous to what we introduced in the bosonic string to discuss the Cardy condition (see eq. (5.212)). The reason can be understood as follows:

$$\begin{aligned}
\text{Tr}\left\{e^{-2\pi tH}\right\} &\overset{\text{Cardy}}{=} \langle B_a|\,e^{-2\frac{\pi}{t}(H+\bar{H})}\,|B_a\rangle = \langle B_a|\,\mathbb{1}\,e^{-2\frac{\pi}{t}(H+\bar{H})}\,\mathbb{1}\,|B_a\rangle\\
&= \sum_{\mathcal{V},\mathcal{V}'}\langle B_a|\mathcal{V}\rangle\,\langle \mathcal{V}|\,e^{-2\frac{\pi}{t}(H+\bar{H})}\,\big|\mathcal{V}'\big\rangle\big\langle \mathcal{V}'\big|B_a\big\rangle\\
&= \sum_{\mathcal{V}}|\langle \mathcal{V}|B_a\rangle|^2\,e^{-\frac{\pi}{t}(h_{\mathcal{V}}+h_{\mathcal{V}})}\,,
\end{aligned}\tag{6.209}$$

where $h_{\mathcal{V}}$ is the weight of $\mathcal{V}$ and where we used a basis where $H$ (and hence $L_0$) is diagonal, which implies $|\mathcal{V}\rangle = \big|\mathcal{V}'\big\rangle$. We decomposed the propagator in all its possible channels:

$$\text{Tr}\left\{e^{-2\pi tH}\right\} = \sum_{\mathcal{V}}\;\;\;\text{[diagram]}\;\;\;.\tag{6.210}$$

We hence realize that $\langle B_a|\mathcal{V}\rangle$ represents the emission of an asymptotic state $\langle \mathcal{V}|$ from the disk of BC $= a$ (see eq. (6.208)).

If we evaluate the R-R potential on a disk having $D(p)$ BC we find

$$\text{[diagram]}\;\sim \mu_p\,,\tag{6.211}$$

which is the R-R charge of a $D(p)$-brane.

On the other hand, for the graviton $G_{\mu\nu}$ (whose vertex operator is written as $\mathcal{V}_G$) we have that

$$\text{[diagram]}\;\sim T_p\,,\tag{6.212}$$

which is the $D(p)$-brane tension, namely its mass per volume unit.

This tells us that

$$\langle D(p)| = T_p\left\langle \text{NS}\,\overline{\text{NS}}\right| + \mu_p\left\langle \text{R}\,\overline{\text{R}}\right|\,,\tag{6.213}$$

where $\left\langle \text{NS}\,\overline{\text{NS}} \right|$ is a state in the NS $\overline{\text{NS}}$ sector, while $\left\langle \text{R}\,\overline{\text{R}} \right|$ is a state in the R $\overline{\text{R}}$ sector.

From a spacetime point of view, the R-R charge of a $D(p)$-brane is obtained by straightorward generalization of Gauss theorem

$$\mu_p = \int_{\mathcal{S}_{8-p}} \star F^{(p+2)} = \int_{\mathcal{S}_{8-p}} F^{(8-p)}, \tag{6.214}$$

being $\mathcal{S}_{8-p}$ an $(8-p)$-sphere surrounding the $D(p)$-brane.

All in all we get the following structure:

$$F^{(p+2)} \quad \xleftrightarrow{\text{Hodge duality}} \quad F^{(8-p)}$$

$$d \uparrow \qquad\qquad\qquad\qquad d \uparrow$$

$$C^{(p+1)} \qquad\qquad\qquad\qquad C^{(7-p)}$$

$$\downarrow \qquad\qquad\qquad\qquad\qquad \downarrow$$

couples to a $D(p)$-brane which is an extended object with electric charge

couples to a $D(6-p)$-brane which is an extended object with magnetic charge

$$\mu_p = \int_{\mathcal{S}_{8-p}} F^{(8-p)} \qquad\qquad \mu_{6-p} = \int_{\mathcal{S}_{p+2}} F^{(p+2)}$$

Polchinski explicitly showed that, if one evaluates separately $\mu_p$ and $\mu_{6-p}$, they are such that (in appropriate units)

$$\mu_p \mu_{6-p} = 2\pi. \tag{6.215}$$

Therefore $D(p)$-branes have the smallest electric and magnetic charge allowed by Dirac quantization condition: they are the fundamental quanta of RR-charge.

In type IIA the fundamental charged objects are therefore

$$\begin{cases} D(0)\text{-brane:} & \langle D(0)| = T_0 \left\langle \text{NS}\,\overline{\text{NS}} \right| + \mu_0 \left\langle \text{R}\,\overline{\text{R}} \right|, \\ \text{anti-}D(0)\text{-brane:} & \left\langle \overline{D(0)} \right| = T_0 \left\langle \text{NS}\,\overline{\text{NS}} \right| - \mu_0 \left\langle \text{R}\,\overline{\text{R}} \right|, \end{cases} \tag{6.216a}$$

$$\begin{cases} D(2)\text{-brane:} & \langle D(2)| = T_2 \left\langle \text{NS}\,\overline{\text{NS}} \right| + \mu_2 \left\langle \text{R}\,\overline{\text{R}} \right|, \\ \text{anti-}D(2)\text{-brane:} & \left\langle \overline{D(2)} \right| = T_2 \left\langle \text{NS}\,\overline{\text{NS}} \right| - \mu_2 \left\langle \text{R}\,\overline{\text{R}} \right|. \end{cases} \tag{6.216b}$$

$$\vdots$$

If we build a $D - \overline{D}$ system with a $D$-brane and a $\bar{D}$-brane on top of each other, their boundary states get summed together and the R $-\overline{\text{R}}$ sector cancels out:

$$\left\langle D(p) - \overline{D(p)} \right| = 2T_p \left\langle \text{NS}\,\overline{\text{NS}} \right|, \tag{6.217}$$

means that the $D - \overline{D}$ system has a vanishing R-R charge. As we will see, this system is unstable, just like an electron-positron pair.

We can also build bosonic-like $D(p)$-branes considering the cases where the corresponding R-R potential is not present, for example in Type IIA:

$$\langle D(1)| = T_1 \left\langle \text{NS}\,\overline{\text{NS}} \right|. \tag{6.218}$$

These $D$-branes are called *non-BPS* and are unstable objects, just like bosonic $D$-branes. In conclusion, type IIA has R-R charged $D(p)$-branes for $p$ even and un-charged (bosonic-like) $D(p)$-branes for $p$ odd.

On the other hand, type IIB has R-R charged $D(p)$-branes for $p$ odd and un-charged (bosonic-like) $D(p)$-branes for $p$ even.

### 6.4.2 *D*-branes and open superstrings

Let us now talk about open strings, whose physics depend on the $D(p)$-branes system we are considering. As we already know, in the open case we have a single holomorphic sector, which splits into NS and R. We thus have to build a boundary SCFT where the gluing condition will be

$$T(z) = \overline{T}(\bar{z}), \tag{6.219a}$$

$$\text{NS)} \quad G(z) = \overline{G}(\bar{z}), \tag{6.219b}$$

$$\text{R)} \quad G(z) = \begin{cases} +\overline{G}(\bar{z}), & \text{for } z = \bar{z} < 0, \\ -\overline{G}(\bar{z}), & \text{for } z = \bar{z} > 0. \end{cases} \tag{6.219c}$$

The gluing condition of the supercurrent in the R case comes from the fact that in $z = \bar{z} = 0$ there is a spin field which generates a branch cut on the real axis from 0 to $\infty$.

The doubling trick for the stress-energy tensor reads as

$$T_{\mathbb{C}} = \begin{cases} T_{\text{UHP}}(z), & \text{for } Im(z) > 0, \\ \overline{T}_{\text{UHP}}(z^*), & \text{for } Im(z) < 0. \end{cases} \tag{6.220}$$

We now define a label in order to distinguish between NS and R:

$$\epsilon = \begin{cases} +1, & \text{for NS}, \\ -1, & \text{for R}. \end{cases} \tag{6.221}$$

Hence we can write the doubling trick for the supercurrent as

$$G_{\mathbb{C}}^{(\epsilon)}(z) = \begin{cases} G_{\text{UHP}}(z), & \text{for } Im(z) > 0, \\ \overline{G}_{\text{UHP}}(z^*), & \text{for } Im(z) > 0, Re(z) < 0, \\ \epsilon \overline{G}_{\text{UHP}}(z^*), & \text{for } Im(z) > 0, Re(z) > 0. \end{cases} \tag{6.222}$$

After a closed path around the origin, we have that

$$G_{\mathbb{C}}^{(\epsilon)}(ze^{2i\pi}) = \epsilon G_{\mathbb{C}}^{(\epsilon)}(z), \tag{6.223}$$

since in the R case there is a branch cut from 0 to $\infty$ on the real axis.

The bosonic current $j^\mu(z)$ has the usual gluing condition

$$j^\mu(z) = \Omega_{\text{A}}^{(j)} \bar{j}^\mu(\bar{z}), \quad \text{for } z = \bar{z}, \tag{6.224}$$

with

$$\Omega_{\text{A}}^{(j)} = \begin{cases} +1, & \text{for A = Neumann}, \\ -1, & \text{for A = Dirichlet}. \end{cases} \tag{6.225}$$

If we then recall that

$$\delta_{\text{SUSY}}^{(\epsilon)} \psi^{\mu(\epsilon)}(z) = j^\mu(z), \tag{6.226}$$

we have that the bosonic gluing condition for $j^\mu(z)$ immediately induces the fermionic gluing condition for $\psi^\mu(z)$, which are

$$\psi^{\mu(\epsilon)}(z) = \begin{cases} \Omega_{\text{A}}^{(\psi)} \bar{\psi}^{\mu(\epsilon)}(\bar{z}), & \text{for } z = \bar{z} < 0, \\ \epsilon \Omega_{\text{A}}^{(\psi)} \bar{\psi}^{\mu(\epsilon)}(\bar{z}), & \text{for } z = \bar{z} > 0, \end{cases} \tag{6.227}$$

where the gluing map $\Omega_{\text{A}}^{(\psi)}$ is the same as $\Omega_{\text{A}}^{(j)}$.

A $D(p)$-brane is a $(X; \psi)$-BCFT where

- $(X^\mu; \psi^\mu)$ have $\Omega = +1$ (Neumann BC) for $\mu = 0, 1, \dots, p$,

- $(X^i; \psi^i)$ have $\Omega = -1$ (Dirichlet BC) for $i = p + 1, \dots, 9$.

These $D(p)$-branes live in a 10-dimensional spacetime codified by closed superstring of type IIA or type IIB.

We now want to see what is the effect of GSO projection on the open strings sector. Given a type IIA/B background, let us consider two boundary states $|D(p_1)\rangle$ and $|D(p_2)\rangle$ in the directions transverse to the light-cone. We then use the Cardy condition and write

$$\langle D(p_1)| e^{-\frac{\pi}{t}(L_0 + \bar{L}_0 - \frac{c}{12})} |D(p_2)\rangle = \mathrm{Tr}_{\mathcal{H}_{12}} \left\{ e^{-2\pi t(L_0 - \frac{c}{24})} \right\}, \qquad (6.228)$$

where $\mathcal{H}_{12}$ is the Hilbert space of the strings stretched between the $D(p_1)$-brane and the $D(p_2)$-brane, transverse to the light-cone. In the following we separately study the cases where $p_1 = p_2 = p$.

### 6.4.2.1 The charged $D(p) - D(p)$ system

Let us consider the $D(p) - D(p)$ branes system, where the two $D(p)$-branes both have the same non-zero R-R charge (hence $p = 2n$ in type IIA and $p = 2n + 1$ in type IIB). The open strings

$$\begin{pmatrix} \phi_{11} & \phi_{12} \\ \phi_{21} & \phi_{22} \end{pmatrix} \qquad (6.229)$$

can have both NS and R sectors. From the Cardy condition it follows that

- in the NS sector the $\mathrm{GSO}_{\mathrm{NS}}^+$ projection leaves the $8_V$ representation and throws away the tachyon:

- in the R sector the $\mathrm{GSO}_{\mathrm{R}}^+$ projection leaves the $8_C$ representation, while the $\mathrm{GSO}_{\mathrm{R}}^-$ projection leaves the $8_S$ representation (this is the $SO(8)$ content, which should be appropriately decomposed following the breaking pattern induced by the $Dp$ boundary conditions).

The double possibility we have in the R sector (namely $\mathrm{GSO}_{\mathrm{R}}^+$ or $\mathrm{GSO}_{\mathrm{R}}^-$) corresponds to the fact that we can have positive R-R charge ($D(p) - D(p)$ system) or negative R-R charge ($\overline{D(p)} - \overline{D(p)}$ system). These GSO projections will be denoted as $\mathrm{GSO}^\pm(+)$, where the superscript refers to the R sector, while the parenthesis refers to the NS sector. The GSO we perform on the open strings are hence

$$\begin{pmatrix} \phi_{11} & \phi_{12} \\ \phi_{21} & \phi_{22} \end{pmatrix} \longrightarrow \begin{pmatrix} \mathrm{GSO}^\pm(+) & \mathrm{GSO}^\pm(+) \\ \mathrm{GSO}^\pm(+) & \mathrm{GSO}^\pm(+) \end{pmatrix}. \qquad (6.230)$$

### 6.4.2.2 BPS condition

Let us now look at the annulus partition function. We have already seen that in the bulk the torus partition is identically vanishing which is the sign that there is space-time supersymmetry. On the boundary, namely for the open string, let us consider a charged $D(p) - D(p)$ system,

where the two $D(p)$-branes are parallel to each other and separated along a Dirichlet direction $y$, at a distance $\Delta y$. Computing the one-loop vacuum amplitude, one finds

$$
\int_0^\infty \frac{dt}{(2t)^2} \mathrm{Tr}^\perp \left[ (-1)^{\mathbb{F}} \frac{1}{2} \left( 1 + (-1)^F \right) e^{-2\pi t (L_0 - \frac{1}{2})} \right]
$$

$$
= \int_0^\infty \frac{dt}{(2t)^2} \left( \frac{1}{2t} \right)^{\frac{p-1}{2}} e^{-\frac{t(\Delta y)^2}{2\pi\alpha'}} \left( \frac{1}{\eta(it)} \right)^8 \left( \frac{\theta_3{}^4 - \theta_4{}^4 - \theta_2{}^4}{\eta^4} \right) (it)
$$

$$
= \frac{1}{4\pi} \int_0^\infty ds \left( \frac{\pi}{s} \right)^{\frac{d_\perp}{2}} e^{-\frac{(\Delta y)^2}{2\alpha' s}} \left( \frac{1}{\sqrt{2}\eta\left(\frac{is}{\pi}\right)} \right)^8 \left( \frac{\theta_3{}^4 - \theta_4{}^4 - \theta_2{}^4}{\eta^4} \right) \left( \frac{is}{\pi} \right)
$$

$$
= \frac{1}{4\pi} \int_0^\infty ds \, \langle D(p) | e^{-s(L_0 + \bar{L}_0 - \frac{c}{12})} | D(p) \rangle
$$

$$
= 0, \tag{6.231}
$$

where we have set $t = \frac{\pi}{s}$ and used the modular properties for the $\theta$ and $\eta$ functions ((6.140a), (6.141a)). This quantity is identically vanishing which means that the open string states stretched between the two identical $D$-branes have a perfectly balanced boson-fermion degeneracy. However in the $s$ variable this has another complementary interpretation. We recall indeed from the bosonic string that the large $s$ behaviour of the integrand is sensible to the massless closed string states that are exchanged between the two $D$-branes. A power series expansion in $s \to \infty$ can isolate the contribution from the exchange of massless states:

$$
\left( \frac{1}{\eta\left(\frac{is}{\pi}\right)} \right)^8 \left( \frac{[\theta_3{}^4 - \theta_4{}^4]_{\mathrm{NSNS}} - [\theta_2{}^4]_{\mathrm{RR}}}{\eta^4} \right) \left( \frac{is}{\pi} \right) \underset{s \to \infty}{\sim} (16_{\mathrm{NSNS}} - 16_{\mathrm{RR}}) + (0)e^{-s} + \dots \tag{6.232}
$$

Considering only the massless exchange, in complete analogy with the bosonic string, we find again the Green function $G(\Delta y) \sim 1/|\Delta y|^{d_\perp - 2}$:

$$
\Big( \overbrace{16}^{\mathrm{NS}} - \overbrace{16}^{\mathrm{R}} \Big) G(\Delta y) = 0. \tag{6.233}
$$

$$
\underset{\substack{=0 \\ \text{matching of bosonic} \\ \text{and fermionic d.o.f.}}}{}
$$

Therefore two charged $D(p)$-branes at a distance $\Delta y$ are attracted by a force coming from the $\mathrm{NS} - \overline{\mathrm{NS}}$ sector (due to the dilaton and the graviton) which is balanced by an equal and opposite force coming from the $\mathrm{R} - \bar{\mathrm{R}}$ sector:

$$
\underbrace{\text{tension}}_{\substack{\text{produces gravitational attraction} \\ \text{between the two } D(p)\text{-branes}}} = \underbrace{\text{R-R charge.}}_{\substack{\text{produces electric repulsion} \\ \text{between the two } D(p)\text{-branes}}} \tag{6.234}
$$

Hence the two $D(p)$-branes do not interact. In supersimmetry, this is called *BPS condition*.

### 6.4.2.3 $\mathcal{N} = 4$ Super Yang-Mills

Let us now consider a stack of $N$ $D(3)$-branes in type IIB theory, where they are R-R charged (and have a field-strength $F^{(5)}$). Given this setting, we have 4 Neumann directions $\mu = 0, 1, 2, 3$ and 6 Dirichlet directions $i = 4, \dots, 10$: therefore these BC break $SO(1,9)$ into $SO(1,3) \otimes SO(6)$.

In order to throw away the tachyon, the GSO projection for open strings will be the following:

$$
\text{Neumann } X^\mu, \psi^\mu \ \longrightarrow \ \begin{cases} \text{NS)} & \text{GSO}^+_{\text{NS}}\,, \\ \text{R)} & \text{GSO}^+_{\text{R}}\,, \end{cases}
$$

$$
\text{Dirichlet } X^i, \psi^i \ \longrightarrow \ \begin{cases} \text{NS)} & \text{GSO}^+_{\text{NS}}\,, \\ \text{R)} & \text{GSO}^+_{\text{R}}\,. \end{cases}
$$

The first state in the NS sector is

$$
A_\mu(P)\psi^\mu_{\frac{1}{2}}\,|0,P\rangle_{\text{NS}} + \lambda_i(P)\psi^\mu_{-\frac{1}{2}}\,|0,P\rangle_{\text{NS}}\,, \tag{6.235}
$$

where $A$ and $\lambda$ are $N \times N$ matrices, since we have a stack of $N$ $D(3)$-branes. The physical constraints are

$$
G_{\frac{1}{2}} \implies P \cdot A = 0\,, \qquad L_0 \implies P^2 = 0\,. \tag{6.236}
$$

Therefore we find $N^2$ gluons (in the adjoint representation of $U(N)$) and $6N^2$ transverse scalars (in the adjoint representation of $U(N)$).

On the other hand, the first state in the R sector in $d = 10$, without considering the GSO projection, is

$$
\chi_\alpha(P)|\alpha,P\rangle_{\text{R}} + \chi_{\dot\alpha}(P)|\dot\alpha,P\rangle_{\text{R}}\,, \tag{6.237}
$$

where $\alpha \in 16_C$ and $\dot\alpha \in 16_S$. When we then perform the GSO projection, the $16_S$ is removed and we are left with a single chiral spinor in $d = 10$. In order to understand how $\alpha$ decomposes in $SO(1,9)$ broken into $SO(1,3) \otimes SO(6)$, we write

$$
\alpha = \left( \pm\frac{1}{2}, \pm\frac{1}{2}, \pm\frac{1}{2}, \pm\frac{1}{2}, \pm\frac{1}{2} \right)\,, \tag{6.238}
$$

and since we are dealing with a left spinor there is an even number of $-1/2$. Therefore we can decompose $\alpha$ as

$$
\alpha = \Big( \underbrace{\pm\frac{1}{2}, \pm\frac{1}{2}}_{a,\dot a}, \underbrace{\pm\frac{1}{2}, \pm\frac{1}{2}, \pm\frac{1}{2}}_{I,\bar I} \Big)\,, \tag{6.239}
$$

which gives rise to the two following cases:

- $\alpha = (a, I)$, where $a$ is a Weyl spinor of $SO(1,3)$ with positive chirality and $I$ is a Weyl spinor of $SO(6)$ with positive chirality,

- $\alpha = (\dot a, \bar I)$, where $\dot a$ is a Weyl spinor of $SO(1,3)$ with negative chirality and $\bar I$ is a Weyl spinor of $SO(6)$ with negative chirality.

The two possible representations $I = 4_C$ and $\bar I = 4_S$ of $SO(6)$ can be seen as the fundamental 4 and the anti-fundamental $\bar 4$ representations of $SU(4)$, since the Dynkin diagram of their algebras are topologically equivalent (see fig. 6.4).
Therefore $\chi_\alpha$ in $d = 10$ can be written as $(\chi^I_\alpha, \chi^{\bar I}_{\dot\alpha})$, which are massless fermions in the fundamental and the anti-fundamental representations of $SU(4)$. They are called *gluinos* (or *gauginos*).

What we found is that in the case of $N$ $D(3)$-branes in type IIB there is an effective theory for the fields

$$
\begin{aligned}
A_\mu &\qquad \text{vectors (spin 1),} \\
\chi^I_\alpha, \chi^{\bar I}_{\dot\alpha} &\qquad \text{gauginos (spin 1/2),} \\
\lambda_i &\qquad \text{scalars (spin 0),}
\end{aligned}
$$

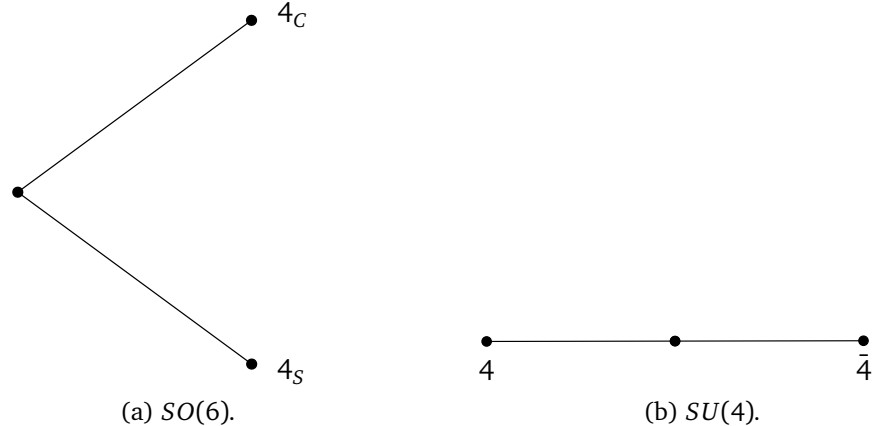

Figure 6.4: The $SO(6)$ and the $SU(4)$ algebra Dynkin diagram are topologically equivalent.

which form the vector multiplet for the $\mathcal{N} = 4$ super Yang-Mills theory. The related action is

$$S_{\mathcal{N}=4}^{\text{SYM}} = \frac{1}{g_s} \int d^4x \, \text{Tr} \Big\{ F_{\mu\nu} F^{\mu\nu} + D_\mu \lambda^i D^\mu \lambda_i + \big[ \lambda_i, \lambda_j \big] \big[ \lambda^i, \lambda^j \big]$$
$$+ \bar{\chi}^{\bar{I}} \not{D} \chi^I + \bar{\chi}^I \big[ \chi^{\bar{I}}, \lambda_i \big] \gamma_i^{I,\bar{I}} \Big\} + \big( \alpha' \text{ corrections} \big), \qquad (6.240)$$

where the term $\big[ \lambda_i, \lambda_j \big] \big[ \lambda^i, \lambda^j \big]$ shows a quartic potential structure for the $\lambda$ field, while the $\gamma$ coefficients are the gamma-matrices of $SO(6)$. Every term of this action can be obtained by computing open strings amplitudes in the $N$ $D(3)$-branes system, as we sketched in the bosonic string example.

Since there is no dimensional parameter inside $S_{\mathcal{N}=4}^{\text{SYM}}$, there is a conformal invariance at the classical level. Moreover, the $\beta$ function of this theory vanishes, which means that it also has a superconformal invariance at the quantum level.

### 6.4.2.4   The $D(p) - \overline{D(p)}$ system

Let us consider a $D(p) - \overline{D(p)}$ branes system, whose total R-R charge is zero. This time the closed string exchange amplitude (or equivalently the one-loop bubble of the stretched string) will be

$$\frac{1}{4\pi} \int_0^\infty ds \, \big\langle \overline{D}(p) \big| e^{-s(L_0 + \bar{L}_0 - \frac{c}{12})} \big| D(p) \big\rangle$$
$$= \frac{1}{4\pi} \int_0^\infty ds \left( \frac{\pi}{s} \right)^{\frac{d_\perp}{2}} e^{-\frac{(\Delta y)^2}{2\alpha' s}} \left( \frac{1}{\sqrt{2} \eta \left( \frac{is}{\pi} \right)} \right)^8 \left( \frac{\theta_3{}^4 - \theta_4{}^4 + \theta_2{}^4}{\eta^4} \right) \left( \frac{is}{\pi} \right)$$
$$= \int_0^\infty \frac{dt}{(2t)^2} \left( \frac{1}{2t} \right)^{\frac{p-1}{2}} e^{-\frac{t(\Delta y)^2}{2\pi\alpha'}} \left( \frac{1}{\eta(it)} \right)^8 \left( \frac{\theta_3{}^4 + \theta_4{}^4 - \theta_2{}^4}{\eta^4} \right) (it). \qquad (6.241)$$

The plus sign in front of $\theta_2$ is there because this is the contribution from the exchange of RR fields. Since the the two RR charges are opposite now this contribution is opposite from the $D - D$ case. After passing to the open string variable $t = \pi/s$ and using the modular transformation we have that under $t \to 1/t$

$$\theta_2/\eta \leftrightarrow \theta_4/\eta \,.$$

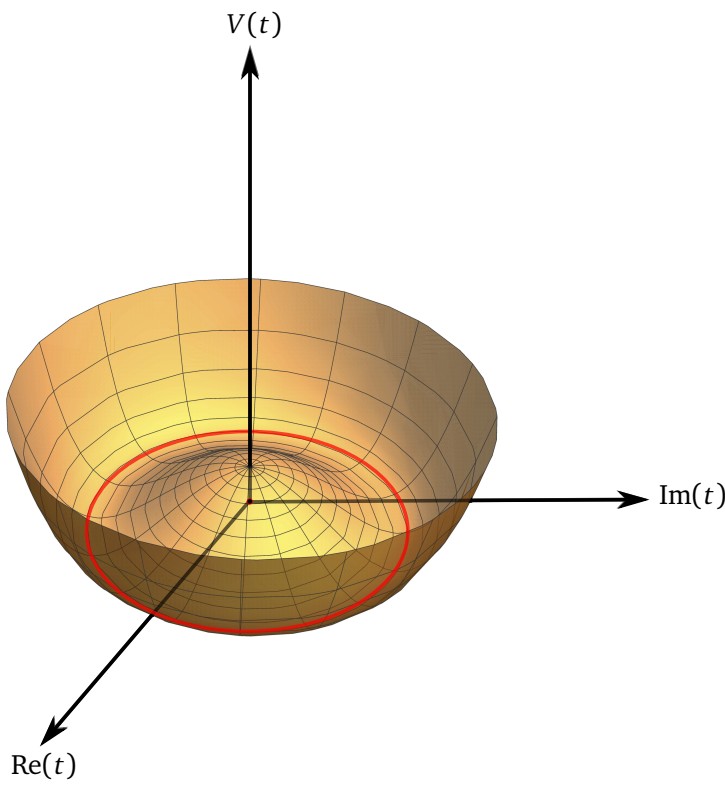

Figure 6.5: The tachyon 'Mexican hat' potential for a $D$-$\bar{D}$ system. The tip of the potential, located in $t = 0$, is the unstable vacuum. The red circle $\mathcal{S}_1^{(t)}$ is the manifold of the minima, where $V(t) = 0$. We hence have a 1-parameter degeneration of the stable vacuum.

Therefore this time we find that the GSO projection for the open string is *opposite* from the one we used in the DD system. In total, from the Cardy condition, this time we find a different GSO projection in the off-diagonal sector, describing open strings stretching from the brane and the anti-brane

$$
\begin{pmatrix} \phi_{11} & \phi_{12} \\ \phi_{21} & \phi_{22} \end{pmatrix} \longrightarrow \begin{pmatrix} \text{GSO}^+(+) & \text{GSO}^+(-) \\ \text{GSO}^-(-) & \text{GSO}^-(+) \end{pmatrix}. \tag{6.242}
$$

Therefore we have a complex open string tachyon in the off-diagonal sector.

From a closed string point of view, we have already discussed that we have

$$
|D(p) - \overline{D(p)}\rangle = 2T_p \left| \text{NS} - \overline{\text{NS}} \right\rangle, \tag{6.243}
$$

and we see that the total R-R charge is zero. On the other hand, from an open string point of view we see a tachyon, hence an unstable vacuum. Therefore we have two complementary informations:

$$
\text{vanishing R-R charge} \iff \text{unstable vacuum.}
$$

This essentially means that, since the $D(p) - \overline{D(p)}$ is not charged, the system is unstable and it will decay. Such a process can be codified by the dynamics of the open string tachyon $t$. It can be shown using string field theory that its effective potential $V(t, \bar{t})$ (where $t \in \mathbb{C}$, since the Chan-Paton is $\begin{pmatrix} 0 & t \\ \bar{t} & 0 \end{pmatrix}$) has a Mexican-hat shape (see fig. 6.5).

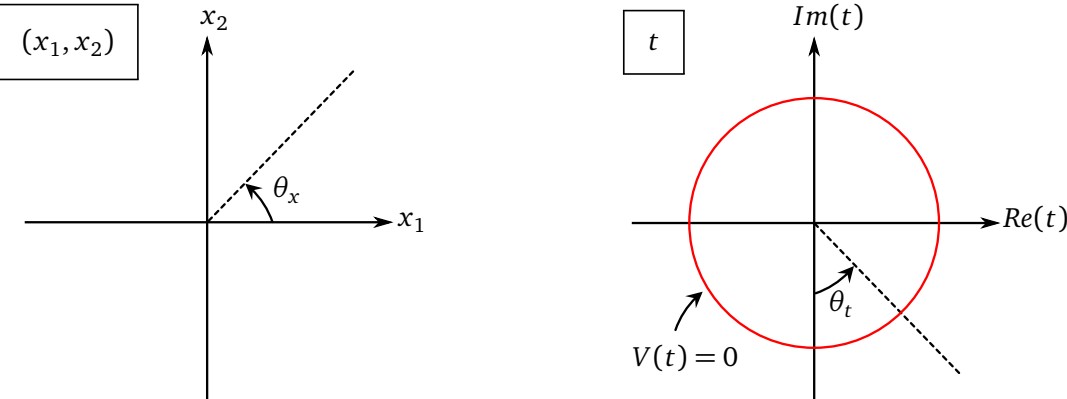

Figure 6.6: On the left, we have the $(x_1, x_2)$ plane where the vortex solution $t(x^1, x^2)$ lives: the perturbative vacuum is in the origin, while the point at infinity is the non-perturbative vacuum (since $t$ acquires vev). On the right, we have the $t$ plane: the red circle is the minima manifold where $V(t) = 0$, namely the 1-parameter degeneration of the stable vacuum, while the origin of the two axis corresponds to the unstable minimum (see figure 6.5).

Let us now write the tachyon as $t = t(x^\mu)$ (where $x^\mu$ are the worldvolume coordinates of the $D(p) - \overline{D(p)}$ system, i.e. $\mu = 0, 1, \ldots, p$). Consider the tachyon equation of motion derived from the effective potential

$$\Box t - V'(t) = 0. \tag{6.244}$$

This equation has several interesting solutions. The simplest is

$$t = 0, \tag{6.245}$$

which corresponds to the initial unstable system. Then we have

$$t = t_* e^{i\theta}, \tag{6.246}$$

where $V'(t_*) = 0$, minimum of the potential. This is a one-parameter family of solutions which describe a constant field configuration. This is a stable vacuum where the $D$-$\bar{D}$ system has dissolved into pure closed string radiation, leaving empty space-time, just like an electron and a positron decay by emitting photons. This is a curious vacuum state for the effective open string description, since there are no more $D$-branes in this vacuum, yet the open string field theory exists. This configuration is called *Tachyon Vacuum*. A detailed analysis using string field theory shows that the energy difference $V(0) - V(t_* e^{i\theta})$ precisely accounts for the total mass of the initial unstable $D - \bar{D}$ system.

It is also interesting to mention the so-called *vortex solution* depending only on two space-like coordinates $x^1, x^2 \in \{x^0, x^1, \ldots, x^p\}$, that is to consider $t(x^1, x^2) = t(\vec{x})$. The gauge fields on the D-branes also play a crucial role, but we will not consider them in our sketchy argument

These vortex solutions are identified by a topological charge, called *winding number*. Its meaning can be grasped as follows. Since the vortex solution has to be normalizable (i.e. the vortex has a finite energy), $t(\vec{x} \to \infty)$ has to be in the minimum of the potential. If $t_*$ is the radius of the minima manifold (see fig. 6.6), and if we write $\vec{x} = |x| e^{i\theta_x}$, what we just said

reads as

$$\lim_{\vec{x} \to \infty} t(\vec{x}) = \lim_{|x| \to \infty} t\left(|x|e^{i\theta_x}\right) = t_* e^{i\theta_t},\tag{6.247}$$

where $\theta_x$ is related to the manifold $\mathcal{S}_1^{(x)}$ of the configurations for $\vec{x} \to \infty$ in the $(x^1, x^2)$ plane, while $\theta_t$ is linked to the manifold $\mathcal{S}_1^{(t)}$ of the stable minima in the $t$ plane.

Thus the generic vortex solution is associated to a map

$$m : \; \mathcal{S}_1^{(x)} \; \longrightarrow \; \mathcal{S}_1^{(t)},\tag{6.248}$$

that is classified by the winding number $n \in \mathbb{Z}$, which counts how many times $\mathcal{S}_1^{(x)}$ winds around $\mathcal{S}_1^{(t)}$. If the winding is performed clockwise, the winding number is positive, and negative otherwise. This is a topological charge, since it cannot be changed by continuous deformations of the map $m$.

As a consequence of all of this, the vortex solution is labelled by the winding number: $t_n(\vec{x})$. The space profile will describe a concentration of mass-energy near $\vec{x} \sim 0$ while for large $\vec{x}$ the solution will be relaxed at the tachyon vacuum where there are no $D$-branes. This solution is therefore describing a charged object which has a world-volume with two dimensions less than the original $D - \bar{D}$ system.

Ashoke Sen conjectured that the vortex solution $t_n(\vec{x})$ represents $n$ coincident $D(p-2)$-branes located at $\vec{x} = 0$. Indeed the $D(p) - \overline{D(p)}$ system has no R-R charge for $F^{(p+2)}$, whereas a stack of $n$ $D(p-2)$-branes has a R-R charge of $n$ given by $F^{(p)}$ and, according to the conjecture, it is exactly the winding number of the tachyonic vortex. A quantitative prediction of this conjecture is that the vortex energy per volume unit is equal to the tension of $n$ $D(p-2)$-branes. This speculation has been numerically proven in string field theory, but a full analytic solution is still lacking.

## 6.5 Bosonization and picture charge (more advanced)

### 6.5.1 Bosonization

#### 6.5.1.1 Spin field

Let's consider type II superstring theory. As we said before, we define the Ramond vacuum as a spin field acting on the $SL(2, \mathbb{C})$-invariant vacuum:

$$|\alpha\rangle_{\mathrm{R}} = S_\alpha(0)|0\rangle,\tag{6.249a}$$

$$|\dot{\alpha}\rangle_{\mathrm{R}} = S_{\dot{\alpha}}(0)|0\rangle.\tag{6.249b}$$

Let us now recall the contractions for the fermions $\psi_I^\pm(z)$ we defined in (A.4):

$$\psi_I^+(z_1)\psi_J^-(z_2) = \frac{\delta_{IJ}}{(z_1 - z_2)},\tag{6.250}$$

$$\psi_I^\pm(z_1)\psi_J^\pm(z_2) = 0.\tag{6.251}$$

We may now consider a logarithmic free chiral boson $H_I(z)$, such that

$$H_I(z_1)H_J(z_2) = -\delta_{IJ}\log(z_1 - z_2),\tag{6.252}$$

with a stress-energy tensor given by

$$T_H(z) = \frac{1}{2} :\partial H \cdot \partial H: (z).\tag{6.253}$$

We can employ it to build the plane waves $e^{\pm iH_I}(z)$, which have the following contractions:

$$e^{+iH_I}(z_1)e^{-iH_J}(z_2) = \frac{\delta_{IJ}}{(z_1 - z_2)}, \tag{6.254a}$$

$$e^{\pm iH_I}(z_1)e^{\pm iH_J}(z_2) = 0. \tag{6.254b}$$

We notice that the contractions of these plane waves are the same as the contractions of the fermionic field $\psi_I^\pm(z)$. We can thus perform a field redefinition called *bosonization*:

$$\psi_I^\pm(z) = e^{\pm iH_I}(z). \tag{6.255}$$

This name comes from the fact that we wrote a fermion as a plane wave of a boson.

We can then use this technique to bosonize the spin field as follows:

$$S_\alpha(z) = e^{i \sum_I s_I H_I}(z), \tag{6.256}$$

where $s_I$ is the so-called spinor weight (recall that we are in $d = 10$):

$$s_I = \left\{ \pm\frac{1}{2}, \pm\frac{1}{2}, \pm\frac{1}{2}, \pm\frac{1}{2}, \pm\frac{1}{2} \right\}. \tag{6.257}$$

According to the definition (A.15), the spinor weight of the left spin field $S_\alpha$ will have an even number of $-1/2$, whereas the spinor weight of the right spin field $S_{\dot\alpha}$ will have an odd number of $-1/2$.

In $d = 10$ the spin field is a product of 5 elementary plane-waves $e^{\pm\frac{i}{2}H_I}$, one for each independent 2-dimensional plane. The weight of $e^{\pm\frac{i}{2}H_I}$ can be easily computed to be $1/8$. This means that

$$h(S_\alpha) = h(S_{\dot\alpha}) = \frac{5}{8}. \tag{6.258}$$

This is the reason why in the definition (6.86) of the Ramond matter+ghost vacuum we took a ghost part $c\mathcal{V}_{3/8}^{(\beta,\gamma)}$ having overall ghost weight $-5/8$: this indeed combines with the matter weight of the spin field, leaving a weight zero physical state. Now we are going to describe the $\beta\gamma$ operator $\mathcal{V}_{3/8}^{(\beta,\gamma)}$.

### 6.5.1.2 Ghost $(\beta, \gamma)$ system

The $\beta$ and $\gamma$ ghosts are (spinorial) bosons and to bosonize them (since they are already bosonic) we need two decoupled CFTs:

$$(\eta, \xi) \oplus \varphi, \tag{6.259}$$

where $\eta$ and $\xi$ are fermions, while $\varphi$ is a boson. With these ingredients we can redefine $\beta$ and $\gamma$ as

$$\beta(z) = e^{-\varphi} \partial \xi(z), \tag{6.260a}$$

$$\gamma(z) = \eta e^\varphi(z). \tag{6.260b}$$

The $(\eta, \xi)$ system (which is kind of a $(b, c)$ system) is such that

$$h(\eta) = 1, \qquad h(\xi) = 0, \tag{6.261}$$

and

$$\xi(z_1)\eta(z_2) = \frac{1}{(z_1 - z_2)} \, . \tag{6.262}$$

The stress-energy tensor of this $(\eta, \xi)$-CFT is

$$T^{(\eta,\xi)}(z) = :(\partial \eta)\xi:(z) - \partial(:\eta\xi:)(z) \, , \tag{6.263}$$

and its central charge turns out to be $c = -2$. Moreover, the normalization is

$$\langle 0| \, \xi_0 \, |0\rangle = 1 \, . \tag{6.264}$$

On the other hand, $\varphi$ is a chiral twisted free boson with logarithmic OPE.

$$\varphi(z_1)\varphi(z_2) = -\log(z_1 - z_2) \, . \tag{6.265}$$

The stress-energy tensor of this $\varphi$-CFT is

$$T^{\varphi}(z) = -\frac{1}{2}(\partial \varphi)(\partial \varphi) - \partial^2 \varphi \, , \tag{6.266}$$

where $\partial^2 \varphi = \partial[\partial \varphi]$ is the twist term (here is where the name "twisted boson" comes from). The central charge turns out to be $c = +13$.

Therefore the overall $\{(\eta, \xi) \oplus \varphi\}$-CFT has a $c = -2 + 13 = +11$ central charge, which is the same as the $(\beta, \gamma)$ system.

We can evaluate

$$h(e^{q\varphi}) = -\frac{q}{2}(q + 2) \, , \tag{6.267}$$

which means that $h = 0$ if and only if

$$e^{q\varphi} = \begin{cases} \mathbb{1} \longrightarrow q = 0 \, , \\ e^{-2\varphi} \longrightarrow q = -2 \, , \end{cases} \tag{6.268}$$

with

$$\langle \mathbb{1} \rangle = 0 \, , \qquad \langle e^{-2\varphi} \rangle = 1 \neq 0 \, . \tag{6.269}$$

This fact is called *anomalous momentum conservation*, and can be explained as follows. If we define the charge

$$Q_{\varphi} = \oint_0 \frac{dz}{2\pi i}(-\partial \varphi) = \oint_0 \frac{dz}{2\pi i} j_{\varphi}(z) \, , \tag{6.270}$$

we have that

$$\left[Q_{\varphi}, e^{q\varphi}\right] = q e^{q\varphi} \, , \tag{6.271}$$

and

$$T_{\varphi}(z_1) j_{\varphi}(z_2) = -\frac{2}{(z_1 - z_2)^3} + \frac{j_{\varphi}(z_2)}{(z_1 - z_2)^2} + \frac{\partial j_{\varphi}(z_2)}{(z_1 - z_2)} + (\text{regular terms}) \, , \tag{6.272}$$

which means that the current $j_{\varphi}$ is not a primary. This has the following effect on BPZ conjugation:

$$Q_{\varphi} |0\rangle = 0 \xrightarrow{\mathcal{I}(z) = -\frac{1}{z}} \langle 0| \left(Q_{\varphi} + 2\right) = 0 \, , \tag{6.273}$$

Table 6.5: The value of the picture charge $p$ for the main fields we are interested in.

| Field | $p$ |
|:---:|:---:|
| $\beta$ | $0$ |
| $\gamma$ | $0$ |
| $\eta$ | $-1$ |
| $\xi$ | $1$ |
| $e^{-q\varphi}$ | $-q$ |
| $|0\rangle_{\mathrm{NS}}$ | $-1$ |
| $|\alpha\rangle_{\mathrm{R}}$ | $-\frac{1}{2}$ |

which implies that

$$\langle \prod_i e^{q_i\varphi}(z_1)\rangle \neq 0 \iff \sum_i q_i = -2\,. \tag{6.274}$$

Thanks to this construction, we can finally write the full expression of the NS and R vacua, namely

$$|0\rangle_{\mathrm{NS}} = e^{-\varphi}(0)\,|0\rangle\,, \qquad |\overset{(\cdot)}{\alpha}\rangle_{\mathrm{R}} = S_{\underset{\alpha}{(\cdot)}} e^{-\frac{\varphi}{2}}(0)\,|0\rangle\,, \tag{6.275}$$

which are essentially what we wrote in (6.84) and (6.86). The NS vacuum has weight $1/2$ while the R vacuum has weight $1$. After dressing them with a $c$-ghost we find that the vacua are respectively $-1/2 = -a_{NS}$ and $0 = -a_R$, which precisely match the lightcone zero point energies, as they should.

### 6.5.2 Picture charge and large Hilbert space

Combining the two CFTs arising from the bosonization of the $(\beta, \gamma)$ system, we can define the current

$$j_p(z) = :\xi\eta:(z) - \partial\varphi(z) = :\xi\eta:(z) + j_\varphi(z)\,, \tag{6.276}$$

the related conserved charge is called *picture charge $p$*. Its values for the fields we are dealing with are written in table 6.5

Now we have three anomalous currents which have to be conserved in correlators. The first is the $(b, c)$ ghost current which fixes the 3 c zero modes (4.180). Then we have anomalous momentum conservation (6.269) and finally we have the $\xi, \eta$ current which fixes a $\xi$. In total the non-vanishing 1-point function is

$$\langle \xi e^{-2\varphi}\frac{1}{2}(\partial^2 c)(\partial c)c(z)\rangle_{\mathrm{LHS}}^{\mathrm{chiral}} = 1\,. \tag{6.277}$$

It has $p = -1$ and $gh = 3$. The label LHS says that the correlator is computed in the so-called *large Hilbert space*. In order to understand what does this mean, we can go back to the field redefinitions (6.260a) and (6.260b) of $\beta$ and $\gamma$. We notice that they do not use the full Hilbert space of the $\{(\eta, \xi) \oplus \varphi\}$-CFT, because the dependence on $\xi$ is all inside $\partial\xi$ (in $\beta$), where the zero-mode $\xi_0$ is excluded. Indeed, since $\xi$ is a weight-zero primary, we can write

$$\xi(z) = \sum_{n=0}^{\infty} \xi_n z^{-n} = \xi_0 + \sum_{n=1}^{\infty} \xi_n z^{-n}\,, \tag{6.278}$$

and realize that inside $\partial \xi$ the zero-mode $\xi_0$ is not present.

We then have to distinguish between

- the small Hilbert space (SHS) where the $(\beta, \gamma)$ system lives and where the zero-mode $\xi_0$ is not present,

- the large Hilbert space (LHS) where the whole $\{(\eta, \xi) \oplus \varphi\}$ system lives and where the zero-mode $\xi_0$ is present.

The LHS is twice as big as the SHS because $\forall \,|\text{state}\rangle \in$ SHS the LHS contains both $|\text{state}\rangle$ and $\xi_0 |\text{state}\rangle$.

String amplitudes and observables will live inside the SHS, since they are build with the $(\beta, \gamma)$ system. Hence, starting from the LHS, it is useful to know how to come back to the physical SHS, and this is simply realized as follows:

$$\boxed{\langle \ldots \rangle_{\text{SHS}} = \langle \xi_0 (\ldots) \rangle_{\text{LHS}} \,.} \tag{6.279}$$

Moreover, in the SHS the correlator (6.277) reads as

$$\boxed{\langle e^{-2\varphi} \frac{1}{2} (\partial^2 c)(\partial c) c(z) \rangle_{\text{SHS}}^{\text{chiral}} = 1 \,.} \tag{6.280}$$

It has $p = -2$ and $gh = 3$.

Let us now study the structure of the LHS a bit more in detail. We can start by considering the zero-modes

$$\xi_0 = \oint_0 \frac{dz}{2\pi i} \frac{1}{z} \xi(z), \qquad \eta_0 = \oint_0 \frac{dz}{2\pi i} \eta(z), \tag{6.281}$$

which obey

$$\boxed{[\eta_0, \xi_0] = 1 \,.} \tag{6.282}$$

Since in the SHS there is no dependence on $\xi_0$, we have that a generic vector in SHS will be annihilated by

$$\frac{\partial}{\partial \xi_0} = \eta_0 = 0 \,. \tag{6.283}$$

Hence we can write

$$\text{SHS} = \mathcal{H}_S = \ker(\eta_0) \,. \tag{6.284}$$

If we now also recall the BRST charge

$$Q_B = \oint_0 \frac{dz}{2\pi i} j_{\text{B}}(z), \tag{6.285}$$

we realize that in the LHS there are two nilpotent operators

$$\boxed{Q_B^2 = 0 \,, \qquad \eta_0^2 = 0 \,,} \tag{6.286}$$

which (anti-)commute with each other:

$$\boxed{[Q_B, \eta_0] = 0 \,.} \tag{6.287}$$

Moreover, both $Q_B$ and $\eta_0$ have empty cohomology:[40]

$$\left[\eta_0, \xi(z)\right] = 1, \qquad \left[Q_B, -c\xi\partial\xi e^{-2\varphi}(z)\right] = 1.$$ 

(6.288)

The operator $\xi(z)$ is called *contracting-homotopy operator* for $\eta$, while $-c\xi\partial\xi e^{-2\varphi}(z)$ is the contracting-homotopy operator for $Q_B$. Notice in particular that the contracting homotopy operator for $Q_B$ explicitly contains $\xi$, which only exists in the LHS. We thus learned that

- in the LHS neither $Q_B$ nor $\eta_0$ have cohomology since they both have a contracting-homotopy operator;

- in the SHS $\eta_0 = 0$ (since we are inside its kernel) and $\xi_0$ is not present: $Q_B$ has no contracting-homotopy operator and has a non-empty cohomology.

The two contracting homotopy operators we have found allow to construct two interesting new operators (by interchanging the role of $Q$ and $\eta$):

$$\mathcal{X}(z) = \left[Q_B, \xi(z)\right],$$ 

(6.289a)

$$\mathcal{Y}(z) = \left[\eta_0, -c\xi\partial\xi e^{-2\varphi}(z)\right] = c\partial\xi e^{-2\varphi}(z).$$ 

(6.289b)

It is immediate to see that

$$\begin{cases} [Q_B, \mathcal{X}(z)] = 0, \\ [\eta_0, \mathcal{X}(z)] = 0, \end{cases} \qquad \begin{cases} [Q_B, \mathcal{Y}(z)] = 0, \\ [\eta_0, \mathcal{Y}(z)] = 0, \end{cases}$$ 

(6.290)

and so $\mathcal{X}, \mathcal{Y} \in$ SHS.

Moreover

- $\mathcal{X}$ has picture $p = +1$ and hence is called *picture raising operator*,

- $\mathcal{Y}$ has picture $p = -1$ and hence is called *picture lowering operator*.

These two operators are one the inverse of the other in the OPE sense:

$$\lim_{z \to 0} \mathcal{X}(z)\mathcal{Y}(0) = 1.$$ 

(6.291)

We now define the zero-mode of $\mathcal{X}$ as

$$\mathcal{X}_0 = \oint_0 \frac{dz}{2\pi i} \frac{1}{z} \mathcal{X}.$$ 

(6.292)

In this way, given a physical state $\phi_p$ of picture $p$ (which, being physical, satisfies $Q_B\phi_p = 0$), we can raise its picture with $\mathcal{X}_0$ and the resulting state will still be physical:

$$\mathcal{X}_0\phi_p = \phi_{p+1}, \qquad Q_B\phi_{p+1} = 0.$$ 

(6.293)

This means that, for a given physical field, we can build an infinite chain of fields that have a different picture charge and yet are still physical:

$$\cdots \underset{\mathcal{Y}}{\overset{\mathcal{X}}{\rightleftarrows}} \phi_{p-1} \underset{\mathcal{Y}}{\overset{\mathcal{X}}{\rightleftarrows}} \phi_p \underset{\mathcal{Y}}{\overset{\mathcal{X}}{\rightleftarrows}} \phi_{p+1} \underset{\mathcal{Y}}{\overset{\mathcal{X}}{\rightleftarrows}} \cdots$$ 

(6.294)

We therefore learn that in superstring theory every physical state is infinitely degenerate because we can take it in any picture we want. While evaluating superstring (tree-level) amplitudes we will have to end up with a total picture of $p = -2$ (recall eq. (6.280)), and this will be the criterion to be employed in order to choose at which picture we want to take our fields.

On top of this, we have the $\mathbb{Z}_2$ degeneracy due to the two possible vertex operators we can use, namely integrated or non-integrated.

---

[40]Given a nilpotent odd operator $d$, if there exists another odd operator $h$ such that $[d, h] = 1$ then, for every $d$-closed state $\psi$, ($d\psi = 0$) we have that $\psi = d(h(\psi))$, so every closed state is exact and $d$ has no cohomology.

### 6.5.3 GSO projection and bosonization

We now want to use what we learned about bosonization to better understand the GSO projection.

First of all, it is useful to recall the field redefinitions of the fermion $\psi$ and of the $\beta$ and $\gamma$ ghosts:

$$\psi_I^{\pm}(z) = e^{\pm i H_I}(z), \tag{6.295a}$$

$$\beta(z) = e^{-\varphi}\partial\xi(z), \tag{6.295b}$$

$$\gamma(z) = \eta e^{\varphi}(z). \tag{6.295c}$$

We can then write a generic state in the NS or in the R sector as a plane wave with generalized momenta $v_I$, $s_I$ (related to the matter $\psi$ part, where $I = 1, \ldots, 5$ since we are in $d = 10$) and $v_6$, $s_6$ (related to the ghost $(\beta, \gamma)$ part):

$$|\text{NS state}\rangle = e^{iv_I H_I + v_6\varphi}, \quad \text{with } v_I, v_6 \in \mathbb{Z}, \tag{6.296a}$$

$$|\text{R state}\rangle = e^{is_I H_I + s_6\varphi}, \quad \text{with } s_I, s_6 \in \mathbb{Z} + \frac{1}{2}. \tag{6.296b}$$

Since, obviously,

$$\text{integer} + \text{integer} = \text{integer}, \tag{6.297a}$$

$$\text{integer} + \text{half-integer} = \text{half-integer}, \tag{6.297b}$$

$$\text{half-integer} + \text{half-integer} = \text{integer}, \tag{6.297c}$$

we have the general OPEs

$$|\text{NS state}\rangle \cdot |\text{NS state}\rangle \approx |\text{NS state}\rangle, \tag{6.298a}$$

$$|\text{NS state}\rangle \cdot |\text{R state}\rangle \approx |\text{R state}\rangle, \tag{6.298b}$$

$$|\text{R state}\rangle \cdot |\text{R state}\rangle \approx |\text{NS state}\rangle. \tag{6.298c}$$

Using GSO projection we can then split the NS sector and the R sector as follows:

$$\text{NS} \longrightarrow \begin{cases} \text{GSO}_{\text{NS}}^+, & \text{where } \sum_I v_I + v_6 \text{ is even,} \\ \text{GSO}_{\text{NS}}^-, & \text{where } \sum_I v_I + v_6 \text{ is odd,} \end{cases} \tag{6.299a}$$

$$\text{R} \longrightarrow \begin{cases} \text{GSO}_{\text{R}}^+, & \text{where } \sum_I s_I + s_6 \text{ is even,} \\ \text{GSO}_{\text{R}}^-, & \text{where } \sum_I s_I + s_6 \text{ is odd.} \end{cases} \tag{6.299b}$$

We immediately see that the $\text{GSO}_{\text{NS}}^+$ sector is closed under OPEs, which means that

$$\text{GSO}_{\text{NS}}^+ \cdot \text{GSO}_{\text{NS}}^+ \approx \text{GSO}_{\text{NS}}^+. \tag{6.300}$$

On the other hand, in the R sector we can fix $s_6$ (which is essentially the picture charge, related to the bosonized ghosts) and the difference between even $\text{GSO}_{\text{R}}^+$ and odd $\text{GSO}_{\text{R}}^-$ reduces to the chirality of the corresponding spacetime fermion $|\{s_I\}\rangle$.

The algebra of these generalized plane waves is

$$e^{ip_I H_I + p_6\varphi}(z_1)e^{iq_I H_I + q_6\varphi}(z_2) = (z_1 - z_2)^{-p_I q_I + p_6 q_6} e^{i(p_I + q_I)H_I + (p_6 + q_6)\varphi}(z_2). \tag{6.301}$$

If we want to ask for locality in the chiral sector (namely the absence of branch cuts) we have to take

$$-p_I q_I + p_6 q_6 \in \mathbb{Z}. \tag{6.302}$$

SciPost Phys. Lect. Notes 90 (2025)

But we have that

$$\text{GSO}^-_{\text{NS}}(z_1) \cdot \text{GSO}^\pm_{\text{R}}(z_2) \approx \frac{1}{\sqrt{z_1 - z_2}}, \tag{6.303a}$$

$$\text{GSO}^-_{\text{R}}(z_1) \cdot \text{GSO}^+_{\text{R}}(z_2) \approx \frac{1}{\sqrt{z_1 - z_2}}. \tag{6.303b}$$

Therefore the minimal consistent sectors where fermions are still in the spectrum (as we would like it to be, since this is the reason we introduced supersimmetry for) are the two following possibilities:

$$\text{GSO}^+_{\text{NS}} \oplus \text{GSO}^+_{\text{R}}, \qquad \text{GSO}^+_{\text{NS}} \oplus \text{GSO}^-_{\text{R}}. \tag{6.304}$$

This is the truncation giving rise to Type II A-B and to spacetime SUSY.

### 6.5.4  $D(p)$-branes and supersymmetry

Let us consider type IIA theory. Since we have two spin fields of opposite chirality, we can build the two following fermionic supercharges:

$$Q_\alpha(z) = S_\alpha e^{-\frac{\varphi}{2}}(z), \tag{6.305a}$$

$$\overline{Q}_{\dot\alpha}(z) = \overline{S}_{\dot\alpha} e^{-\frac{\overline{\varphi}}{2}}(\overline{z}). \tag{6.305b}$$

Using the contractions

$$\underline{S_\alpha(z_1)S_\beta(z_2)} = \frac{1}{\sqrt{2}} \frac{(\Gamma_\mu)_{\alpha\beta}}{(z_1 - z_2)^{\frac{3}{4}}} \psi^\mu(z_2), \tag{6.306a}$$

$$\underline{S_\alpha(z_1)S_{\dot\beta}(z_2)} = \frac{C_{\alpha\dot\beta}}{(z_1 - z_2)^{\frac{5}{4}}} + \frac{1}{2} \frac{\Gamma_\mu\Gamma_\nu}{(z_1 - z_2)^{\frac{1}{4}}} :\psi^\mu\psi^\nu: (z_2), \tag{6.306b}$$

$$\underline{e^{-\frac{\varphi}{2}}(z_1)e^{-\frac{\varphi}{2}}(z_2)} = \frac{1}{(z_1 - z_2)^{\frac{1}{4}}} e^{-\varphi}(z_2), \tag{6.306c}$$

we can evaluate

$$\underline{Q_\alpha(z_1)Q_\beta(z_2)} = \frac{1}{\sqrt{2}} \frac{(\Gamma_\mu)_{\alpha\beta}}{(z_1 - z_2)} \psi^\mu e^{-\varphi}(z_2), \tag{6.307}$$

where the $\varphi$ term comes from $e^{-\varphi/2}e^{-\varphi/2} = e^{-\varphi}$, and hence the total picture charge of the result is $-1/2 - 1/2 = -1$. We would have expected something like $Q_\alpha(z_1)Q_\beta(z_2) \approx \Gamma_\mu j^\mu$, but instead of the momentum generator $j^\mu$ we have found its representative at picture (-1), namely $\psi^\mu e^{-\varphi}$:

$$\mathcal{X}_0(\psi^\mu e^{-\varphi}) = j^\mu. \tag{6.308}$$

We thus learn one important point of the RNS string: *Space time supersymmetry is only realized up to picture changing.*

On the other hand, if we want to consider two supercharges of different chirality in the same holomorphic sector we get

$$\underline{Q_\alpha(z_1)Q_{\dot\beta}(z_2)} = \frac{C_{\alpha\dot\beta}}{(z_1 - z_2)^{\frac{3}{2}}} e^{-\varphi}(z_2) + \frac{1}{2} \frac{\Gamma_\mu\Gamma_\nu}{(z_1 - z_2)^{\frac{1}{2}}} :\psi^\mu\psi^\nu: (z_2). \tag{6.309}$$

It has branch cuts, but we got rid of this bad situation through GSO projection, that forbids the combination $Q_\alpha(z_1)Q_{\dot\beta}(z_2)$.

In general, we have

- type IIA, with $Q_\alpha$ and $\bar{Q}_{\dot{\beta}}$,

- type IIB, with $Q_\alpha$ and $\bar{Q}_\beta$.

If we now want to study $D(p)$-branes, we have to give a set of gluing conditions. For the bosonic currents we already know how to proceed, but the question now is whether it is possible or not to consistently glue the two chiral parts of the supercharge $Q$ at the boundary.

If we choose type IIB theory, we can write a plausible gluing condition

$$Q_\alpha(z) = \mathscr{P}_\alpha{}^\beta \bar{Q}_\beta(\bar{z}), \quad \text{for } z = \bar{z}, \tag{6.310}$$

and try to find the gluing map $\mathscr{P}_\alpha{}^\beta$.

If we take $M = (\{\mu\}, \{i\})$ (where $\{\mu\}$ are the Neumann indices and $\{i\}$ are the Dirichlet indices), we can write the OPE $Q_\alpha(z_1) Q_\beta(z_2)$. The left-hand side is simply given by (6.307):

$$\underline{Q_\alpha(z_1) Q_\beta(z_2)} = \frac{1}{\sqrt{2}} \frac{(\Gamma_M)_{\alpha\beta}}{(z_1 - z_2)} \psi^M e^{-\varphi}(z_2). \tag{6.311}$$

The right-hand side can be written using the gluing condition (6.310) and reads as

$$\underline{\mathscr{P}_\alpha{}^\gamma \bar{Q}_\gamma(\bar{z}_1) \mathscr{P}_\beta{}^\delta \bar{Q}_\delta(\bar{z}_2)} = \frac{1}{\sqrt{2}} \frac{(\Gamma_M)_{\alpha\beta}}{(z_1 - z_2)} \Omega^{(\psi)M}{}_N \bar{\psi}^N e^{-\bar{\varphi}}. \tag{6.312}$$

We wrote the usual gluing map (6.225) as a matrix:

$$\Omega_N^M = \begin{pmatrix} \delta_\nu^\mu & 0 \\ 0 & -\delta_j^i \end{pmatrix} \longleftrightarrow \Omega_A = \begin{cases} +1, & \text{for A = Neumann,} \\ -1, & \text{for A = Dirichlet.} \end{cases} \tag{6.313}$$

In this notation, the usual bosonic and fermionic gluing conditions read as

$$\begin{cases} j^M(z) = \Omega_N^M \bar{j}^N(\bar{z}), & \text{for } z = \bar{z}, \\ \psi^M(z) = \Omega_N^M \bar{\psi}^N(\bar{z}), & \text{for } z = \bar{z}. \end{cases} \tag{6.314}$$

Therefore, if we want the left-hand side and the right-hand side to be compatible, we have to impose

$$\mathscr{P}_\alpha{}^\gamma \mathscr{P}_\beta{}^\delta (\Gamma_N)_{\gamma\delta} = (\Gamma_M)_{\alpha\beta} \Omega_M^N, \tag{6.315}$$

which can be written as

$$\mathscr{P} \Gamma_N \mathcal{C}^{-1} \mathscr{P}^T = \Omega_N^M \Gamma_M \mathcal{C}^{-1}, \tag{6.316}$$

where $T$ denotes transposition.

It can then be shown that, for a given $D(p)$-brane, the relation (6.316) is solved by the gluing map

$$\mathscr{P}_p^\pm = \pm \prod_{i=p+2}^{9} (\Gamma_i \Gamma), \tag{6.317}$$

being $\Gamma$ the chirality matrix. Thus we find two solutions. However in type IIB (which is the setting we chose) $\mathscr{P}$ must have indices of the same chirality, namely $\mathscr{P} = \mathscr{P}_\alpha{}^\beta$. Since every $\Gamma$ matrix changes the chirality, this implies that $p$ must be odd. Hence we found a consistent gluing condition for the supercharge only for $D(2p-1)$-branes.

On the other hand, if we are in type IIA we have that the gluing condition is

$$Q_\alpha(z) = \mathscr{P}_\alpha{}^{\dot{\beta}} \bar{Q}_{\dot{\beta}}(\bar{z}), \quad \text{for } z = \bar{z}, \tag{6.318}$$

and the gluing map has indices of opposite chirality (i.e. $\mathscr{P} = \mathscr{P}_\alpha{}^{\dot\beta}$). This implies that $p$ inside $\mathscr{P}_p^\pm$ is even and hence we have a consistent gluing condition for the supercharge only for $D(2p)$-branes.

What we just found teaches us that in type IIA or type IIB we have charged $D(p)$-branes preserving half of the spacetime SUSY. The general gluing condition for the supercharge is

$$\boxed{Q(z) = \mathscr{P}\overline{Q}(\tilde{z}), \quad \text{for } z = \bar{z},} \tag{6.319}$$

with

$$\boxed{\mathscr{P}_p^\pm = \pm \prod_{i=p+2}^{9} (\Gamma_i \Gamma).} \tag{6.320}$$

The $\pm$ inside $\mathscr{P}_p^\pm$ distinguishes between $D(p)$-branes and $\overline{D(p)}$-branes. This fact gives us a geometrical intuition which can be grasped us follows. Let us recall that the difference between $D(p)$-branes and $\overline{D(p)}$-branes is the sign of the R-R charge. Its value is related to the integration on the worldvolume $\mathcal{M}_{p+1}$ of the $p$-dimensional $D(p)$-brane of the corresponding R-R potential $C^{(p+1)}$. But since $C^{(p+1)}$ is a volume-form for the manifold $\mathcal{M}_{p+1}$, it defines an orientation:

$$\int_{\mathcal{M}_{p+1}} C^{(p+1)} \longrightarrow \begin{cases} > 0 \implies \text{positive orientation} \implies D(p)\text{-brane,} \\ < 0 \implies \text{negative orientation} \implies \overline{D(p)}\text{-brane.} \end{cases}$$

Hence a $\overline{D(p)}$-brane is just a $D(p)$-brane with opposite orientation, and this is consistent with the $\pm$ ambiguity inside $\mathscr{P}_p^\pm$.

We can then talk about preserved SUSY. We have that

- a $D(p)$-brane preserves $Q + \mathscr{P}_p\overline{Q}$,

- a $\overline{D(p)}$-brane preserves $Q - \mathscr{P}_p\overline{Q}$.

They are both $1/2$ BPS because they preserve only half of the supersimmetry.

If we have many $D(p)$-branes (or many $\overline{D(p)}$-branes), SUSY is still preserved. On the other hand, a $D(p)-\overline{D(p)}$ branes system breaks SUSY completely (in fact the system is unstable with a tachyon, with consequences discussed in 6.4.2.2).

Finally, let us consider a $D(p) - D(q)$ branes system. Here there is a certain number of Neumann-Dirichlet (ND) directions, along which a twisting in the boundary conditions occurs. From what we have just seen we have that

- the $D(p)$-brane preserves $Q + \mathscr{P}_p\overline{Q}$,

- the $D(q)$-brane preserves $Q + \mathscr{P}_q\overline{Q}$, which can be written as

$$Q + \mathscr{P}_q\overline{Q} = Q + \left(\mathscr{P}_p\mathscr{P}_p^{-1}\right)\mathscr{P}_q\overline{Q} = Q + \mathscr{P}_p\left(\prod_{i\in\text{ND}}\Gamma_i\right)\overline{Q}. \tag{6.321}$$

Therefore the preserved supersymmetry must satisfy

$$\mathscr{P}_p^{-1}\mathscr{P}_q\overline{Q} = \overline{Q}, \tag{6.322}$$

which requires that the matrix $\mathscr{P}_p^{-1}\mathscr{P}_q$ has some eigenvalues equal to 1.

Since $(\Gamma_i\Gamma_{i+1})^2 = -1$, the matrix $\Gamma_i\Gamma_{i+1}$ has eigenvalues $\pm i$, which is not good. We thus see that the only possibility to have the proper eigenvalues is to have a difference between the number of Neumann directions on the two branes that is 4 or a multiple of 4, namely

$$p - q = 4n, \quad \text{with } n \in \mathbb{N}. \tag{6.323}$$

These systems preserve 1/4 of SUSY and therefore are called 1/4 BPS. Different combinations (for example the $D(p) - D(p + 2)$ system) break SUSY completely.

## 6.6 Effective superstring theories

There are 5 superstring theories in 10 dimensions, having space-time supersymmetry. We only studied the Type II A and Type IIB. There are however 3 more perturbative string theories that can be constructed. The list of the 5 superstring is as follows

- Type IIA:

    - closed and open strings ($D$-branes) in $d = 10$;
    - 32 supercharges;
    - 2 real Weyl spinors (left+right) in $d = 10$.

- Type IIB:

    - closed and open strings ($D$-branes) in $d = 10$;
    - 32 supercharges;
    - 2 real Weyl spinors (left+left) in $d = 10$.

- Type I:

    - Unoriented closed and open strings in $d = 10$;
    - 16 supercharges;
    - 32 space-filling D9-branes whose RR charge is annihilated by an orientifold making the string unoriented.

- Heterotic $SO(32)$:

    - closed strings in $d = 10$.
    - 16 supercharges;
    - no D-branes
    - $SO(32)$ gauge symmetry realized through a self-dual compactification

- Heterotic $E_8 \times E_8$:

    - closed strings in $d = 10$.
    - 16 supercharges;
    - no D-branes
    - $E_8 \times E_8$ gauge symmetry realized through a self-dual compactification

We can take a low energy limit of each one of these superstring theories. This will give rise to an effective theory ($+\alpha'$ corrections).

- Type IIA $\xrightarrow{\text{low energy}}$ Type IIA SUGRA:

    - $16_S + 16_C$ SUSY;
    - effective action given by eq. (6.167);
    - related to Type IIB SUGRA by $T$-duality.

- Type IIB $\xrightarrow{\text{low energy}}$ Type IIB SUGRA:

    - $16_C + 16_C$ SUSY;
    - effective action given by eq. (6.168);
    - related to Type IIA SUGRA by $T$-duality.

- Type I $\xrightarrow{\text{low energy}}$ Type I SUGRA:

    - $16_C$ SUSY;
    - coupled to $d = 10$ super Yang-Mills with $SO(32)$ gauge group.

- Heterotic $SO(32) \xrightarrow{\text{low energy}}$ Type I SUGRA:

    - $16_C$ SUSY;
    - coupled to $d = 10$ super Yang-Mills with $SO(32)$ gauge group.

- Heterotic $E_8 \times E_8 \xrightarrow{\text{low energy}}$ Type I SUGRA:

    - $16_C$ SUSY;
    - coupled to $d = 10$ super Yang-Mills with $E_8 \times E_8$ gauge group.

Moreover for type I supergravities we have that

$$g_s^{\text{heterotic}} = \frac{1}{g_s^{\text{type I}}}, \tag{6.324}$$

which means that the heterotic weak coupling is the same as the type I strong coupling. This is a duality between a perturbative regime and a non-perturbative regime. As we will see later on, this is called $S$-duality. This relation between the heterotic and the type I superstring will be part of the M-theory picture (see fig. 7.1).

# 7 Introduction to string dualities

## 7.1 Circle compactification and *T*-Duality

### 7.1.1 Compactification on a circle

When we perform a *compactification* on a circle, it means that we shift from a $\mathbb{R}^{1,d}$ spacetime to a $\mathcal{M}^{1,d}$ spacetime defined as

$$\mathcal{M}^{1,d} = \mathbb{R}^{1,d-1} \times \mathcal{S}_R^1, \tag{7.1}$$

where $\mathcal{S}_R^1$ is a circle of radius $R$.

If we call $y$ the coordinate on the circle (and $x^\mu$ the coordinates on $\mathbb{R}^{1,d-1}$, with $\mu = 0, 1, \ldots, d-1$), we have that the periodicity on $\mathcal{S}_R^1$ turns into the following condition:

$$y \sim y + 2\pi R. \tag{7.2}$$

### 7.1.2 Compactification of a scalar

Let us consider, on the compactified spacetime $\mathcal{M}^{1,d}$, a scalar field $\phi(x^\mu, y)$. If we choose it to be massless, it will satisfy the massless Klein-Gordon equation

$$\Box \phi(x^\mu, y) = 0, \tag{7.3}$$

with $\Box = \Box_x + \partial_y^2$. We can Fourier expand the field in harmonics on the circle as

$$\phi(x^\mu, y) = \sum_{n \in \mathbb{Z}} \phi_n(x^\mu) e^{in\frac{y}{R}}, \tag{7.4}$$

and the reality of the field $\phi(x^\mu, y)$ turns into the condition

$$\phi_{-n}^*(x^\mu) = \phi_n(x^\mu). \tag{7.5}$$

The EOMs are then

$$\Box \phi(x^\mu, y) = \sum_{n \in \mathbb{Z}} \left( \Box_x - \frac{n^2}{R^2} \right) \phi_n(x^\mu) e^{in\frac{y}{R}} = 0. \tag{7.6}$$

The $\phi_n(x^\mu)$ fields can be seen as $\infty$ scalar fields in $\mathbb{R}^{1,d-1}$ whose EOMs are:

$$\left( \Box_x - \frac{n^2}{R^2} \right) \phi_n(x^\mu) = 0. \tag{7.7}$$

Therefore they have a mass:

$$m_n^2 = \frac{n^2}{R^2}. \tag{7.8}$$

We thus found that we can consider a massless scalar field in the compactified space $\mathbb{R}^{1,d-1} \times \mathcal{S}_R^1$ or, equivalently, an *infinite tower* of massive scalar fields (of increasing mass) living in $\mathbb{R}^{1,d-1}$, which are called *Kaluza-Klein fields*.

### 7.1.3 Compactification of a gauge field

The story becomes more interesting when we consider the fate of spacetime fields having a non-trivial tensor structure. Think for example of an Abelian gauge field $A_M(x^\mu, y)$. Upon compactification the vector index $M$ will split into $(\mu, y)$ and therefore the $\mathbb{R}^{1,d-1}$ observer will perceive a gauge field $A_\mu$ with its KK tower, i.e.

$$A_\mu(x, y) = A_\mu^0(x) + \sum_{n \neq 0} A_\mu^n(x) e^{iny/R}, \tag{7.9}$$

plus a scalar $\phi \equiv A_y$ with again its KK tower, namely

$$A_y(x, y) = A_y^0(x) + \sum_{n \neq 0} A_y^n(x) e^{iny/R}. \tag{7.10}$$

If we only focus on the KK zero mode $n = 0$ we can see that the electromagnetism in $\mathbb{R}^{1,d-1} \times S_R^1$ becomes precisely electromagnetism in $\mathbb{R}^{1,d-1}$ plus the action for a scalar field $\phi(x) \equiv A_y^0(x)$

$$-\frac{1}{4} \int d^d x \, dy \, F_{MN} F^{MN} = -\frac{1}{4} \int d^d x \left( F_{\mu\nu} F^{\mu\nu} + \partial_\mu \phi \, \partial^\mu \phi + \text{KK fields} \right). \tag{7.11}$$

It is not difficult to see that, by the same mechanism, a $p$-form field $A_{M_1 \cdots M_p}$ will give rise to a $p$-form field $A_{\mu_1 \cdots \mu_p}$, together with a $(p-1)$ form field $B_{\mu_1 \cdots \mu_{p-1}} \equiv A_{\mu_1 \cdots \mu_{p-1}, y}$.

### 7.1.4 Compactification of gravity

The circle compactification of General Relativity is even more interesting, although a little bit more technical. Without entering in the explicit computation let us give the main result. Upon circle compactification, we can express the total metric tensor $g_{MN}$ in $\mathbb{R}^{1,d-1}$ language as

$$g_{MN} = \begin{pmatrix} g_{\mu\nu} - e^{2\phi} A_\mu A_\nu & e^{2\phi} A_\mu \\ e^{2\phi} A_\mu & e^{2\phi} \end{pmatrix}. \tag{7.12}$$

Then it is a classic result that, in the zero mode KK sector the Einsten-Hilbert action in $d + 1$ dimension becomes Einstein-Hilbert in $d$ dimension, coupled to electromagnetism and a dilaton like field $\phi$

$$\int d^d x \, dy \, R^{(d+1)} = \int d^d x \left( R^{(d)} - \frac{1}{4} e^{2\phi} F_{\mu\nu} F^{\mu\nu} - 2 e^{-\phi} \Box e^\phi + \text{KK fields} \right). \tag{7.13}$$

### 7.1.5 Closed bosonic string

It is now time to see what happens when we compactify string theory on a circle.

#### 7.1.5.1 Momentum and winding

Let us now consider the bosonic string theory. If we perform the compactification (7.1), the free boson gets split into

- a free boson $X^\mu(w, \bar{w})$ living in $\mathbb{R}^{1,d-1}$ (with $\mu = 0, 1, \ldots, d-1$),

- a free boson $Y(w, \bar{w})$ living on the circle $\mathcal{S}_R^1$.

In order to implement the periodic constraint (7.2), we set

$$Y^{(R)}(w, \bar{w}) = Y(w, \bar{w}) + A\sigma, \tag{7.14}$$

where $Y$ is the un-compactified free boson. Then we fix $A$ so that

$$\sigma \to \sigma + 2\pi \implies y \to y + 2\pi R\omega, \tag{7.15}$$

where $\omega \in \mathbb{Z}$ is called *winding* and counts how many times the string wraps around the circle $\mathcal{S}_R^1$. It's immediate to see that we have to impose

$$A = R\omega. \tag{7.16}$$

Going to the usual complex coordinates on the complex plane

$$z = e^w, \qquad w = t + i\sigma, \tag{7.17}$$

and hence

$$Y(z, \bar{z}) = Y_0 - \frac{i}{2}\alpha'\left(P_y + R\omega/\alpha'\right)\log z - \frac{i}{2}\alpha'\left(P_y - R\omega/\alpha'\right)\log \bar{z} + \text{(oscillators)}. \tag{7.18}$$

The $R\omega$ terms are the only parts that make this $Y$ boson different from the usual free boson. Moreover, the center of mass momentum $P_y$ along the compactified direction has to be quantized as

$$P_y = \frac{n}{R}, \quad \text{with } n \in \mathbb{Z}. \tag{7.19}$$

Therefore the final solution is

$$Y(z, \bar{z}) = Y_0 - \frac{i}{2}\alpha'\left(\frac{n}{R} + \frac{R\omega}{\alpha'}\right)\log z - \frac{i}{2}\alpha'\left(\frac{n}{R} - \frac{R\omega}{\alpha'}\right)\log \bar{z} + \text{(oscillators)}, \tag{7.20}$$

with

- $\frac{n}{R}$ Kaluza-Klein momentum (with $n \in \mathbb{Z}$),

- $R\omega$ winding (with $\omega \in \mathbb{Z}$).

It is convenient to split $Y(z, \bar{z})$ into its left and right-moving parts:

$$Y(z, \bar{z}) = Y_L(z) + Y_R(\bar{z}), \tag{7.21}$$

$$\begin{cases} Y_L(z) = \frac{Y_0 - c}{2} - \frac{i}{2}\left(\alpha'\frac{n}{R} + R\omega\right)\log z + i\sqrt{\frac{\alpha'}{2}}\sum_{m \neq 0}\frac{\alpha_m^y}{m}z^{-m}, \\ Y_R(\bar{z}) = \frac{Y_0 + c}{2} - \frac{i}{2}\left(\alpha'\frac{n}{R} - R\omega\right)\log \bar{z} + i\sqrt{\frac{\alpha'}{2}}\sum_{m \neq 0}\frac{\tilde{\alpha}_m^y}{m}\bar{z}^{-m}. \end{cases} \tag{7.22}$$

Then we consider the usual $U(1)$ current as

$$\begin{cases} j(z) = i\sqrt{\frac{2}{\alpha'}}\partial Y_L(z) = \sum_{m \in \mathbb{Z}}\alpha_m^y z^{-m-1}, \\ \bar{j}(\bar{z}) = i\sqrt{\frac{2}{\alpha'}}\bar{\partial} Y_R(\bar{z}) = \sum_{m \in \mathbb{Z}}\tilde{\alpha}_m^y \bar{z}^{-m-1}, \end{cases} \tag{7.23}$$

and

$$\begin{cases} \alpha_0^y = \oint_0 \frac{dz}{2\pi i}j(z) = \sqrt{\frac{\alpha'}{2}}\left(\frac{n}{R} + \frac{R\omega}{\alpha'}\right) \equiv \sqrt{\frac{\alpha'}{2}}p_L, \\ \tilde{\alpha}_0^y = \oint_0 \frac{d\bar{z}}{2\pi i}\bar{j}(\bar{z}) = \sqrt{\frac{\alpha'}{2}}\left(\frac{n}{R} - \frac{R\omega}{\alpha'}\right) \equiv \sqrt{\frac{\alpha'}{2}}p_R. \end{cases} \tag{7.24}$$

The compactification process does not affect the OPEs, but it only changes the definition of the zero-modes in the compactified direction.

We define momentum and winding operators as

$$\hat{\mathcal{P}} = \frac{1}{\sqrt{2\alpha'}}\left(\alpha_0^y + \tilde{\alpha}_0^y\right), \tag{7.25}$$

$$\hat{\mathcal{W}} = \frac{1}{\sqrt{2\alpha'}}\left(\alpha_0^y - \tilde{\alpha}_0^y\right), \tag{7.26}$$

where the winding has been normalized to have the same dimensions as the momentum. The states that carry momentum are called momentum (or Kaluza-Klein) modes and are plane waves of the type

$$\left|\frac{n}{R}\right\rangle = e^{i\frac{n}{R}(Y_L+Y_R)}(z,\bar{z})\,|0\rangle\,\Big|_{z=\bar{z}=0}, \tag{7.27}$$

with

$$\hat{\mathcal{P}}\left|\frac{n}{R}\right\rangle = \frac{n}{R}\left|\frac{n}{R}\right\rangle. \tag{7.28}$$

The states that carry winding are called winding modes and are "twisted" plane waves of the type

$$\left|\frac{\omega R}{\alpha'}\right\rangle = e^{i\frac{\omega R}{\alpha'}(Y_L-Y_R)}(z,\bar{z})\,|0\rangle\,\Big|_{z=\bar{z}=0}, \tag{7.29}$$

with

$$\hat{\mathcal{W}}\left|\frac{\omega R}{\alpha'}\right\rangle = \frac{\omega R}{\alpha'}\left|\frac{\omega R}{\alpha'}\right\rangle. \tag{7.30}$$

It is like we had two position operators, i.e. $Y_L + Y_R$ and its dual $Y_L - Y_R$

$$\begin{cases} Y(z,\bar{z}) &= Y_L(z) + Y_R(\bar{z}) \\ &= Y_0 - i\frac{\alpha'}{2}\hat{\mathcal{P}}\log z\bar{z} - i\frac{\alpha'}{2}\hat{\mathcal{W}}\log\frac{z}{\bar{z}} + i\sqrt{\frac{\alpha'}{2}}\sum_{m\neq 0}\left(\frac{\alpha_m^y}{m}z^{-m} + \frac{\tilde{\alpha}_m^y}{m}\bar{z}^{-m}\right), \\ \widetilde{Y}(z,\bar{z}) &= Y_L(z) - Y_R(\bar{z}\bar{z}) \\ &= C - i\frac{\alpha'}{2}\hat{\mathcal{W}}\log z\bar{z} - i\frac{\alpha'}{2}\hat{\mathcal{P}}\log\frac{z}{\bar{z}} + i\sqrt{\frac{\alpha'}{2}}\sum_{m\neq 0}\left(\frac{\alpha_m^y}{m}z^{-m} - \frac{\tilde{\alpha}_m^y}{m}\bar{z}^{-m}\right). \end{cases} \tag{7.31}$$

Let us now consider a generic state having both momentum and winding. Its vertex operator can be written as

$$\begin{aligned} \mathcal{V}_{n,\omega}(z,\bar{z}) &= e^{i\frac{n}{R}Y}(z,\bar{z})\cdot e^{i\frac{R\omega}{\alpha'}\widetilde{Y}}(z,\bar{z}) \\ &= e^{i\left(\frac{n}{R}+\frac{R\omega}{\alpha'}\right)Y_L}(z)\cdot e^{i\left(\frac{n}{R}-\frac{R\omega}{\alpha'}\right)Y_R}(\bar{z}). \end{aligned} \tag{7.32}$$

We can then evaluate

$$\left[L_0, \mathcal{V}_{n,\omega}(z,\bar{z})\right] = \frac{\alpha'}{4}\left(\frac{n}{R} + \frac{R\omega}{\alpha'}\right)^2 \mathcal{V}_{n,\omega}(z,\bar{z}), \tag{7.33a}$$

$$\left[\bar{L}_0, \mathcal{V}_{n,\omega}(z,\bar{z})\right] = \frac{\alpha'}{4}\left(\frac{n}{R} - \frac{R\omega}{\alpha'}\right)^2 \mathcal{V}_{n,\omega}(z,\bar{z}). \tag{7.33b}$$

### 7.1.5.2 Spectrum

We just saw that the Hilbert space of a compactified free boson $Y$ is the same as the non-compactified exception made for zero-modes, that carry momentum and winding. The compactification radius $R$ defines a 1-parameter family of string backgrounds, since for every $R$ we have a different $Y$-CFT.

If we then want to look at the spectrum of the whole $X + Y$ free boson, we first have to study the Virasoro constraints, which are the following.

SciPost Phys. Lect. Notes 90 (2025)

- The mass-shell is

$$
\begin{aligned}
L_0 + \bar{L}_0 &= \frac{1}{2}\left(\alpha_0^2 + \tilde{\alpha}_0^2\right) + N + \tilde{N} \\
&= 2 \\
&= \left(\frac{\alpha' P^2}{2} + \frac{\alpha'}{2}\left(\frac{n}{R}\right)^2 + \frac{\alpha'}{2}\left(\frac{R\omega}{\alpha'}\right)^2\right) + N + \tilde{N},
\end{aligned}
\tag{7.34}
$$

where $P$ is the momentum along the non-compactified $\mu$ directions. Using the dispersion relation in the uncompactified directions

$$
\frac{\alpha' P^2}{2} = -\frac{\alpha' m^2}{2},
\tag{7.35}
$$

we get the mass-shell condition

$$
m^2 = \underbrace{\left(\frac{n}{R}\right)^2}_{\substack{\text{Kaluza-Klein}\\\text{momentum}}} + \underbrace{\left(\frac{R\omega}{\alpha'}\right)^2}_{\text{winding}} + \underbrace{\frac{2(N+\tilde{N})}{\alpha'}}_{\text{oscillators}} - \underbrace{\frac{4}{\alpha'}}_{\substack{\text{zero point}\\\text{energy}}},
\tag{7.36}
$$

with $n, \omega \in \mathbb{Z}$ are momentum and the winding.

- The level matching is

$$
\begin{aligned}
L_0 - \bar{L}_0 &= \frac{1}{2}\left(\alpha_0^2 - \tilde{\alpha}_0^2\right) + N - \tilde{N} \\
&= 0 \\
&= \frac{1}{2}\left((\alpha_0^y)^2 - (\tilde{\alpha}_0^y)^2\right) + N - \tilde{N} \\
&= \frac{1}{2}\left(4\frac{\alpha'}{2}\frac{n}{R}\frac{R\omega}{\alpha'}\right) + N - \tilde{N} \\
&= \frac{1}{2}(2n\omega) + N - \tilde{N},
\end{aligned}
\tag{7.37}
$$

since only compactified zero-modes give contribution to $\alpha_0^2 - \tilde{\alpha}_0^2$. Therefore the constraint is

$$
n\omega = \tilde{N} - N.
\tag{7.38}
$$

Notice that the simultaneous presence of winding and KK momentum in a state results in an unbalance of the oscillator level.

In the following we analyze the closed bosonic string spectrum at the massless level.

■ **Generic radius**

For a generic radius $R$, in order to have a massless state we have to impose $n = \omega = 0$ (and hence in this case we will not find anything peculiar). If we denote the global spacetime index as $M = (\mu, y)$, we can write the generic massless state as

$$
G_{MN}(P)\alpha_{-1}^M \tilde{\alpha}_{-1}^N |0, P\rangle.
\tag{7.39}
$$

Since the effect of the compactification is

$$
SO(1,d) \longrightarrow \underbrace{SO(1,d-1)}_{\mu},
\tag{7.40}
$$

we can split the polarization tensor as

$$G_{MN} \longrightarrow \begin{cases} G_{\mu\nu}, \\ G_{\mu y}, G_{y\nu}, \\ G_{yy}. \end{cases} \tag{7.41}$$

If we now recall from ( 3.125) that the field content of this state is made of a graviton $h$, a Kalb-Ramond $B$ and a dilaton $\Phi$, we have that the compactification effect is the following:

$$h_{MN} \longrightarrow h_{\mu\nu}, A_\mu = h_{\mu y}(\text{and } A_\nu = h_{y\nu}), \varphi = h_{yy}, \tag{7.42a}$$

$$B_{MN} \longrightarrow B_{\mu\nu}, B_\mu = B_{\mu y}(\text{and } B_\nu = B_{y\nu}), \tag{7.42b}$$

$$\Phi \longrightarrow \Phi. \tag{7.42c}$$

Since $A_\mu$ and $B_\mu$ are gauge bosons, we find a gauge symmetry $U(1) \otimes U(1)$ of the Yang-Mills type.

### ■ Self-dual radius

If we look at the mass formula (7.36) we see that something special happens when

$$\frac{1}{R} = \frac{R}{\alpha'}. \tag{7.43}$$

The radius that satisfies this equation is called *self-dual radius*:

$$\boxed{R = \sqrt{\alpha'}.} \tag{7.44}$$

It is the radius such that the compactification scale is the same as the string scale.

At the self-dual radius, the mass-shell becomes

$$\alpha' m^2 = n^2 + \omega^2 + 2(N + \tilde{N}) - 4. \tag{7.45}$$

We find new massless states, the simplest of which are the following.

- Scalar massless states with $N = \tilde{N} = 0$ and

$$\begin{cases} n = \pm 2, & \omega = 0, \\ n = 0, & \omega = \pm 2. \end{cases} \tag{7.46}$$

They are scalars because there are no oscillators and thus there is no Lorentz index.

- We can get extra massless vectors with

$$\begin{cases} n = \pm 1, & \omega = \pm 1, \\ n = \pm 1, & \omega = \mp 1. \end{cases} \tag{7.47}$$

The level matching imposes

$$\begin{cases} 1 = N - \tilde{N}, \\ -1 = N - \tilde{N}, \end{cases} \implies \begin{cases} N = 1, \tilde{N} = 0, \\ N = 0, \tilde{N} = 1. \end{cases} \tag{7.48}$$

So the total state will contain an unpaired oscillator. The $Y$ part of the states will be written as

$$\begin{cases} |\pm 1, \pm 1\rangle = e^{\pm \frac{2i}{\sqrt{\alpha'}} Y_L}(z) |0\rangle \Big|_{z=0}, \\ |\pm 1, \mp 1\rangle = e^{\pm \frac{2i}{\sqrt{\alpha'}} Y_R}(\bar{z}) |0\rangle \Big|_{\bar{z}=0}. \end{cases} \tag{7.49}$$

We hence have purely holomorphic or purely anti-holomorphic plane waves.

In total we can build the following massless vectors.

· In the holomorphic sector

$$A_{L,\mu}^{(0)} j^\mu \bar{j}^y e^{iP\cdot X}(z,\bar{z}),$$ (7.50a)

$$A_{L,\mu}^{(+)} j^\mu e^{\frac{2i}{\sqrt{\alpha'}}Y_R} e^{iP\cdot X}(z,\bar{z}),$$ (7.50b)

$$A_{L,\mu}^{(-)} j^\mu e^{\frac{-2i}{\sqrt{\alpha'}}Y_R} e^{iP\cdot X}(z,\bar{z}).$$ (7.50c)

· In the anti-holomorphic sector

$$A_{R,\mu}^{(0)} j^y \bar{j}^\mu e^{iP\cdot X}(z,\bar{z}),$$ (7.51a)

$$A_{R,\mu}^{(+)} e^{\frac{2i}{\sqrt{\alpha'}}Y_L} \bar{j}^\mu e^{iP\cdot X}(z,\bar{z}),$$ (7.51b)

$$A_{R,\mu}^{(-)} e^{\frac{-2i}{\sqrt{\alpha'}}Y_L} \bar{j}^\mu e^{iP\cdot X}(z,\bar{z}).$$ (7.51c)

For generic values of the radius we only had (7.50a) and (7.51a), and the arising gauge symmetry was $U(1)\otimes U(1)$. Now that, at the self dual radius, we build $3+3$ massless vectors, we have a gauge symmetry given by $SU(2)_L \otimes SU(2)_R$. This can be understood as follows. If we define

$$j^0 = j^y,$$ (7.52a)

$$j^+ = e^{\frac{2i}{\sqrt{\alpha'}}Y_L},$$ (7.52b)

$$j^- = e^{-\frac{2i}{\sqrt{\alpha'}}Y_L},$$ (7.52c)

and

$$j^1 = \frac{1}{\sqrt{2}}\left(j^+ + j^-\right) = \sqrt{2}\cos(2Y),$$ (7.53a)

$$j^2 = \frac{1}{\sqrt{2}}\left(j^+ - j^-\right) = \sqrt{2}\sin(2Y),$$ (7.53b)

$$j^3 = j^0,$$ (7.53c)

we find the algebra

$$j^a(z_1)j^b(z_2) = \frac{\delta^{ab}}{(z_1-z_2)^2} + i\varepsilon^{ab}{}_c \frac{j^c(z_2)}{(z_1-z_2)},$$ (7.54)

which has an additional term than the usual OPE (4.126). The structure constant of $SU(2)$ $\varepsilon^{ab}{}_c$ appears. This is a non-Abelian current algebra called *Kac-Moody algebra*. This mechanism of symmetry enhancement is the same that generates the non-Abelian gauge symmetry in the heterotic string.

### 7.1.5.3 *T*-duality

Let us look at the constraints that generate the spectrum

$$\begin{cases} m^2 = \left(\frac{n}{R}\right)^2 + \left(\frac{R\omega}{\alpha'}\right)^2 + \frac{2(N+\tilde{N})}{\alpha'} - \frac{4}{\alpha'}, \\ n\omega = N - \tilde{N}. \end{cases}$$ (7.55)

If we also consider the momentum modes $e^{i\frac{n}{R}Y}$ and the winding modes $e^{i\frac{R\omega}{\alpha'}\tilde{Y}}$, we realize that there is a symmetry between momentum and winding:

$$\begin{cases} R \longrightarrow \frac{\alpha'}{R}\,, \\ n \longrightarrow \omega\,, \\ Y \longrightarrow \tilde{Y}\,. \end{cases} \tag{7.56}$$

This is called *T-duality* and its meaning is that the string cannot distinguish between a compactification on the circle $\mathcal{S}_R^1$ or on the circle $\mathcal{S}_{\alpha'/R}^1$.

In QFT in order to understand which is the compactification radius it is sufficient to measure the masses of Kaluza-Klein modes (see eq. (7.8)), but strings also have winding modes and the theory cannot discriminate between the two.

It is also clear why we called "self-dual" the radius $R = \sqrt{\alpha'}$: under $T$-duality it is mapped to itself.

As a final remark, we notice that in QFT we could, in principle, compactify on a circle of radius $R = 0$. However, in string theory "small" compactifications of $R < \sqrt{\alpha'}$ are dual to "bigger" ones of $R > \sqrt{\alpha'}$. Thus the inequivalent compactifications can be taken to be the ones having $R > \sqrt{\alpha'}$, and the string is thus (again) protected from the quantities smaller than the string scale.

### 7.1.6 Open bosonic string and $D(p)$-branes

Under $T$-duality we have that

$$Y \to \tilde{Y} \implies \begin{cases} Y_L \to Y_L\,, \\ Y_R \to -Y_R\,. \end{cases} \tag{7.57}$$

Let us then take a free compactified boson $Y$ with boundary condition

$$j^y(z) = \Omega^{(j^y)}\bar{j}(\bar{z})\,, \quad \text{for } z = \bar{z}\,, \tag{7.58}$$

where the gluing map is the usual one (see eq. (6.225)).

Under $T$-duality the gluing condition turns into

$$j^y(z) = -\Omega^{(j^y)}\bar{j}(\bar{z})\,, \quad \text{for } z = \bar{z}\,. \tag{7.59}$$

Therefore the effect of $T$-duality on the boundary conditions is

$$\text{Neumann} \xleftrightarrow{\text{T-duality}} \text{Dirichlet.} \tag{7.60}$$

This means that

$$\begin{array}{ccc} D(p)\text{-brane wrapping around } S_R^1 & \xleftrightarrow{\text{T-duality}} & D(p-1)\text{-brane transverse to } S_{\alpha'/R}^1\,, \\ \text{(i.e. } S_R^1 \text{ with Neumann BC)} & & \end{array}$$

$$\begin{array}{ccc} D(p)\text{-brane transverse to } S_R^1 & \xleftrightarrow{\text{T-duality}} & D(p+1)\text{-brane wrapping around } S_{\alpha'/R}^1\,. \\ \text{(i.e. } S_R^1 \text{ with Dirichlet BC)} & & \end{array}$$

### 7.1.7 Superstring

Let us now consider, for example, a type IIA superstring theory. We may compactify $X^y = Y$ (and hence $M = (\mu, y)$ with $\mu = 0, 1, \ldots, 8$). Its supersymmetric partner is $\psi^y$, with

$$\begin{cases} \delta_{\text{SUSY}}\sqrt{2/\alpha'}\,Y = \psi^y\,, \\ \delta_{\text{SUSY}}\psi^y = j^y\,. \end{cases} \tag{7.61}$$

This implies that under $T$-duality we have that

$$Y_R \xrightarrow{\text{$T$-duality}} -Y_R \qquad \implies \qquad \bar{\psi}^y \xrightarrow{\text{$T$-duality}} -\bar{\psi}^y \,, \tag{7.62}$$

both in NS or R sector. In particular, in the R sector

$$\bar{\psi}_0^y \xrightarrow{\text{$T$-duality}} -\bar{\psi}_0^y \,. \tag{7.63}$$

In the $\bar{\text{R}}$ sector have considered the world-sheet fermion-counting operator

$$(-1)^{\bar{F}} = \bar{\Gamma}_{11} e^{-i\pi \sum_{n\geq 1} \bar{\psi}_{-n}\cdot\bar{\psi}_n} \,. \tag{7.64}$$

Therefore, remembering that $\psi_0^\mu \sim \Gamma^\mu$, we have

$$\bar{\Gamma}_{11} e^{-i\pi \sum_{n\geq 1} \bar{\psi}_{-n}\cdot\bar{\psi}_n} \xrightarrow{\text{$T$-duality}} -\bar{\Gamma}_{11}(-1)^{-i\pi \sum_{n\geq 1} \bar{\psi}_{-n}\cdot\bar{\psi}_n} \,. \tag{7.65}$$

So the net effect of $T$-duality is to change the chirality chosen by the GSO projection of the anti-holomorphic R sector. This means that, when compactified, Type-IIA and Type-IIB are, at the end of the day, the *same* theory:

$$\begin{array}{c}\text{type IIA}\\ \text{compactified on } S_R^1\end{array} \xleftrightarrow{\text{$T$-duality}} \begin{array}{c}\text{type IIB}\\ \text{compactified on } S_{\alpha'/R}^1\end{array}. \tag{7.66}$$

Let us now study how $T$-duality affects the R-R potentials. Since

$$X_R^{\;9} \xrightarrow{\text{$T$-duality}} -X_R^{\;9} \,, \tag{7.67}$$

$T$-duality is a parity operation on $X_R^{\;9}$ (the anti-holomorphic sector).

Moreover, under parity spinors transform as

$$\tilde{S}_\alpha \xrightarrow{\text{$T$-duality}} \left(\Gamma_{11}\Gamma^9\tilde{S}\right)_\alpha \,, \tag{7.68}$$

and therefore a fermionic bilinear transforms as

$$S^T \mathcal{C}\Gamma^j\tilde{S} \xrightarrow{\text{$T$-duality}} \begin{cases} S^T \mathcal{C}\Gamma^j\Gamma^9\tilde{S}, & \text{for } j \neq 9\,, \\ S^T \mathcal{C}\tilde{S}, & \text{for } j = 9\,. \end{cases} \tag{7.69}$$

Therefore we have that

$$F^{(p)} S^T \mathcal{C}\Gamma^{\mu_1\dots\mu_9}\tilde{S} \xrightarrow{\text{$T$-duality}} \begin{cases} F^{(p-1)}, & \text{for } 9 \in \{\mu_1,\dots,\mu_9\}\,, \\ F^{(p+1)}, & \text{for } 9 \notin \{\mu_1,\dots,\mu_9\}\,. \end{cases} \tag{7.70}$$

This leads to

$$\begin{array}{cc} \text{type IIA/B} & \text{type IIB/A} \\ F^{(p+1)} \text{ (i.e. } F^{\mu_1\dots\mu_p 9}) \xrightarrow{\text{$T$-duality}} & F^{(p)} \text{ (i.e. } F^{\mu_1\dots\mu_p}), \\ F^{(p-1)} \text{ (i.e. } F^{\mu_1\dots\mu_{p-1}}) \xrightarrow{\text{$T$-duality}} & F^{(p+1)} \text{ (i.e. } F^{\mu_1\dots\mu_{p-1} 9}). \end{array}$$

If we have a $D(2p)$-brane in type IIA:

$$D(2p)\text{-brane} \xrightarrow{\text{$T$-duality}} \begin{cases} D(2p-1)\text{-brane} & \text{\scriptsize if the $D(2p)$-brane was} \\ & \text{\scriptsize wrapped around } \mathcal{S}^1, \\ D(2p+1)\text{-brane} & \text{\scriptsize if the $D(2p)$-brane was not} \\ & \text{\scriptsize wrapped around } \mathcal{S}^1. \end{cases} \tag{7.71}$$

Moreover it can be shown that $T$-duality transforms the dilaton as follows:

$$e^{-\Phi} \xrightarrow{\ T\text{-duality}\ } e^{-\Phi'} = e^{-\Phi} \frac{R}{\sqrt{\alpha'}}, \tag{7.72}$$

and hence at the self-dual radius the dilaton is also self-dual.

For the string constant we thus have that

$$g_s = e^{\langle \Phi \rangle} \xrightarrow{\ T\text{-duality}\ } g_s' = g_s \frac{\sqrt{\alpha'}}{R}, \tag{7.73}$$

which tells us that the perturbation theory under $T$-duality stays the same, up to a constant factor (given by the compactification radius) that however does not change the topological series expansion. This means that $T$-duality is a perturbative duality, which maps perturbative amplitudes into perturbative amplitudes. As we will see this is the only string duality that is perturbative on both sides.

$T$-duality can be extended to more complicated compactification geometries. For example a rather straightforward generalization is to consider toroidal compactifications. A much less trivial generalization is to consider compactifications of the ten dimensional superstrings on the so-called *Calabi-Yau manifolds* which are six dimensional, and thus give rise to EFT in four dimensions. Just as a circle of radius $R$ is dual to a circle of radius $\alpha'/R$, every Calabi-Yau has a dual and the generalized $T$-duality between the two compactifications is called *Mirror Simmetry*.

## 7.2  *S*-duality

The so-called $S$-duality relates the weak coupling regime (where there is a worldsheet description) and the strong coupling regime (where we cannot perform the perturbative topological series anymore, and thus we have no worldsheet description).

Let us start from the effective theory of type IIB superstring, i.e. with the type IIB SUGRA, whose action is (recall eq. (6.168))

$$S_{\text{IIB}} = \frac{1}{2K_{10}^2} \int d^{10}x \sqrt{-G} \left\{ e^{-2\Phi} \left( R + 4(\partial \Phi)^2 - \frac{1}{2} |H_{(3)}|^2 \right) + \dots \right\}. \tag{7.74}$$

Let us then perform the following field redefinition on the metric:

$$G_{MN}^{(\text{E})} = e^{-\frac{1}{2}\Phi} G_{MN}^{(\text{s})}, \tag{7.75}$$

where the E stands for "Einstein frame", while the s stands for "string frame". Doing this we get

$$\begin{aligned} S_{\text{IIB}} = &\frac{1}{2K_{10}^2} \int d^{10}x \sqrt{-G^{(E)}} \left\{ R - \frac{\partial_\mu \tau \partial^\mu \bar{\tau}}{2(\text{Im}(\tau))^2} - \frac{1}{2} \frac{\left| G_{(3)} \right|^2}{\text{Im}(\tau)} - \frac{1}{4} \left| F_{(5)} \right| \right\} \\ &+ \frac{1}{8iK_{10}^2} \int \frac{1}{\text{Im}(\tau)} G_{(4)} \wedge G_{(3)} \wedge \overline{G}_{(3)}, \end{aligned} \tag{7.76}$$

where

- $\tau$ is a spacetime scalar in $d = 10$ defined as

$$\tau = C_{(0)} + i e^{-\Phi}, \tag{7.77}$$

  with $C_{(0)}$ the R-R potential that couples to the $D(-1)$-brane and $\Phi$ the NS-NS dilaton,

- $G_{(3)}$ is defined as

$$G_{(3)} = F_{(3)} - ie^{-\Phi}H_{(3)}, \tag{7.78}$$

where $F_{(3)} = dC_{(2)}$ is the R-R field-strength that couples to the $D(1)$-string, while $H_{(3)} = dB_{(2)}$ is the Kalb-Ramond field-strength that couples to the $F(1)$-string (i.e. the fundamental string already mentioned above, see eq. (3.154)),

- $F_{(5)} = dC_{(4)}$ is the field-strength that couples to the $D(3)$-brane.

This effective action $S_{\text{IIB}}$ is invariant under $SL(2,\mathbb{R})$ transformations, i.e. maps of the type

$$\tau \quad \longrightarrow \quad \frac{a\tau + b}{c\tau + d}. \tag{7.79}$$

This is a global field transformation which also comes with the following map:

$$\begin{pmatrix} C_{(2)} \\ B_{(2)} \end{pmatrix} \quad \longrightarrow \quad \begin{pmatrix} a & b \\ c & d \end{pmatrix}\begin{pmatrix} C_{(2)} \\ B_{(2)} \end{pmatrix}. \tag{7.80}$$

If we select the specific map (which we will call $S$) such that

$$S: \quad \tau \longrightarrow -\frac{1}{\tau}, \tag{7.81}$$

and if we choose a background where $C_{(0)} = 0$, recalling that (see eq. 5.140) and that $\tau = C_{(0)} + ie^{-\Phi}$, we can write

$$S: \quad g_s \longrightarrow \frac{1}{g_s}. \tag{7.82}$$

We can now *conjecture* that this is a symmetry of the full Type IIB String Theory. The first step is to understand how the other string theory parameter ($\alpha'$) is transformed by $S$-duality. This can be obtained by noting that under the S transformation we have

$$B_2 \rightarrow -C_2, \tag{7.83}$$
$$C_2 \rightarrow B_2. \tag{7.84}$$

Since $C_2$ couples to the D1 brane in the same way as $B_2$ couples to the fundamental F1 string, this strongly suggests that under S duality a $D(1)$-brane should turn into a F1-string. Let us remember that the tension of a $D(p)$-brane is given by

$$T_p = \frac{1}{g_s}\left(\frac{1}{2\pi}\right)^p (\alpha')^{-\frac{p+1}{2}}. \tag{7.85}$$

We now demand that under $S$-duality the tension of the $D(1)$-brane becomes the tension of the fundamental string

$$T_{D(1)} = \frac{1}{2\pi g_s \alpha'} \quad \overset{S}{\longleftrightarrow} \quad \frac{1}{2\pi\alpha'}. \tag{7.86}$$

This is realized if and only if

$$S: \quad \alpha' \longrightarrow g_s\alpha'. \tag{7.87}$$

Let us then see what happens to the other objects of IIB string theory

- If we look at the $D(3)$-brane we find that it is self-dual:

$$D(3) \quad \overset{S}{\longleftrightarrow} \quad D(3), \tag{7.88}$$

since

$$T_{D(3)} = \frac{1}{(2\pi)^3 g_s (\alpha')^2} \xleftrightarrow{\ S\ } \frac{g_s}{(2\pi)^3 (g_s \alpha')^2} = \frac{1}{(2\pi)^3 g_s (\alpha')^2} = T_{D(3)}. \tag{7.89}$$

This is perfectly consistent with the fact the the 5-form is also self-dual

$$F_{(5)} \xleftrightarrow{\ S\ } F_{(5)}. \tag{7.90}$$

- If we look at the $D(5)$-brane we have that it is exchanged with the $NS(5)$-brane,[41] namely

$$D(5) \xleftrightarrow{\ S\ } NS(5)\text{-brane}, \tag{7.91}$$

since

$$T_{D(5)} = \frac{1}{(2\pi)^5 g_s (\alpha')^3} \xleftrightarrow{\ S\ } \frac{1}{(2\pi)^5 g_s^2 (\alpha')^3}. \tag{7.92}$$

The $NS(5)$ brane is the magnetic dual of the fundamental string, just like the $D(5)$ brane is the magnetic dual of the $D(1)$-brane. Therefore it makes perfect sense that under $S$-duality the $D$-string is sent to the $F$-string and the $D(5)$ is sent to the $NS(5)$.

The $S$-duality conjecture is the answer to the question of how to characterize the strong coupling limit of type IIB superstring. Indeed, when the string coupling constant is not infinitesimal, string perturbation theory breaks down and we don't know anymore how to compute amplitudes. The situation is similar to QCD where, when we go down in energy, the coupling constant grows and non-perturbative physics enter the game. $S$-duality is telling us something quite astonishing: at strong coupling the d.o.f. of string theory reshuffle and reorganize in such a way as to form the *same* theory at weak coupling, where perturbation theory is again valid!

## 7.3 M-theory

Let us now talk about the strong coupling limit in type IIA superstring. We may start considering a $D(0)$-brane, whose tension (and mass) is

$$T_{D(0)} = \frac{1}{g_s \sqrt{\alpha'}}. \tag{7.93}$$

The $D(0)$-branes are R-R charged (under $F_{(2)}$), they do not decay nor they interact with one another. Thus we can take a bound state of $n \in \mathbb{N}$ coincident $D(0)$-branes, whose tension is

$$T_{nD(0)} = n T_{D(0)}. \tag{7.94}$$

The mass$^2$ of this object is

$$(m_{nD(0)})^2 = \left(\frac{n}{g_s \sqrt{\alpha'}}\right)^2, \tag{7.95}$$

which resembles the Kaluza-Klein tower:

$$m_n^2 = \left(\frac{n}{R}\right)^2. \tag{7.96}$$

---

[41]The $NS(5)$-brane is a gravity soliton, i.e. a non-perturbative object. To be more precise, its origin lies in a $(5+1)$-dimensional Kalb-Ramond-charged black brane solution of type II SUGRA (where such a black brane is the higher-dimensional analogue of a charged black hole). Since the Kalb-Ramond field and the graviton constituting this solution are in the NS-NS sector, we call it $NS(5)$-brane.

We immediately see that (7.96) equals (7.95) for

$$R = g_s \sqrt{\alpha'}. \tag{7.97}$$

This means that looking at the type IIA spectrum we find a Kaluza-Klein tower, which reveals the presence of a compactified dimension. This calls on the $d = 11$ SUGRA, whose action is

$$\begin{aligned}
S_{d=11} =& \frac{1}{2K_{11}^2} \int d^{11}x \sqrt{-G} \left( R - \frac{1}{2} \left| dA_{(3)} \right|^2 \right) \\
&+ \frac{1}{2K_{11}^2} \left( -\frac{1}{6} \right) \int A_{(3)} \wedge F_{(4)} \wedge F_{(4)}.
\end{aligned} \tag{7.98}$$

The type IIA SUGRA is the compactification of the $d = 11$ SUGRA on a circle:

$$\mathcal{M}^{11} = \mathbb{R}^{1,9} \times \mathcal{S}_R^1. \tag{7.99}$$

We have that

$$K_{11}^2 = 2\pi R K_{10}^2, \tag{7.100}$$

since

$$\int d^d X$$

$$\downarrow$$

$$\text{compactification:} \quad \underbrace{\mathcal{M}^d}_{X} = \underbrace{\mathbb{R}^{1,d-2}}_{x} \times \underbrace{\mathcal{S}_R^1}_{y}$$

$$\downarrow$$

$$\int d^d X = \int d^{d-1}x \int dy = 2\pi R \int d^{d-1}x, \tag{7.101}$$

and hence

$$\frac{1}{K_d^2} \int d^d X (\cdots)$$

$$\downarrow$$

$$\frac{2\pi R}{K_{d-1}^2} \int d^{d-1}x(\cdots). \tag{7.102}$$

If we look at the metric in $d = 11$ we have that

$$G_{MN} \longrightarrow \begin{cases} G_{\mu\nu} (\mu, \nu = 0, 1, \ldots, 9), & d = 10 \text{ metric in type IIA} & \text{(NS-NS)}, \\ G_{\mu\,10} \to C_\mu, & \text{vector in type IIA} & \text{(R-R)}, \\ G_{10\,10} \to e^{2\Phi}, & \text{scalar in type IIA} & \text{(NS-NS)}. \end{cases}$$

Moreover, we have a potential with 3 indices in $d = 11$:

$$A_{MNP} \longrightarrow \begin{cases} A_{\mu\nu\rho} (\mu, \nu, \rho = 0, 1, \ldots, 9), & d = 10 \text{ 3-form in type IIA} & \text{(R-R)}, \\ A_{\mu\nu\,10} \to B_{\mu\nu}, & \text{Kalb-Ramond in type IIA} & \text{(NS-NS)}. \end{cases}$$

What we are saying is that the strong coupling limit of the type IIA superstring is the UV-completion of the $d = 11$ SUGRA:

|  | Weak Coupling | | Strong Coupling |
|---|---|---|---|
| UV | Type IIA Superstring | | M-theory? |
| | $\alpha' \to 0 \downarrow$ | | $? \uparrow$ |
| IR | type IIA SUGRA | $\xleftarrow[\text{compactification on } \mathcal{S}_1]{}$ | $d = 11$ SUGRA |

This M-theory is the hypothetical UV completion of the $d = 11$ SUGRA (whose coupling constant is the Plank mass in $d = 11$, i.e. $m_{11}$), but not much is known about it.

The fact that there is a 3-form, which couples to something that has 2 dimensions, suggests the existence of a 2-dimensional object that we may call $M(2)$-brane. It is the fundamental object of M-theory and its tension should be naturally expressed as

$$T_{M(2)} = \frac{(m_{11})^3}{(2\pi)^2}. \tag{7.103}$$

It is possible to show that

$$K_{10} = \frac{(4\pi\alpha')^2}{2\sqrt{\pi}} g_s, \tag{7.104a}$$

$$K_{11}^2 = 2\pi R K_{10}^2 = \frac{1}{4\pi}\left(\frac{2\pi}{m_{11}}\right)^2. \tag{7.104b}$$

Thus we find that the 11-th dimension is compactified on a circle of radius

$$R_{11} = g_s \sqrt{\alpha'}, \tag{7.105}$$

and that the Plank mass in $d = 11$ is

$$m_{11} = g_s^{-\frac{1}{3}} \frac{1}{\sqrt{\alpha'}}. \tag{7.106}$$

These are the two scales of M-theory, which are mapped into the two d.o.f. of string theory, i.e. $\alpha'$ and $g_s$:

$$\alpha' = \frac{l_{11}^3}{R_{11}}, \qquad g_s = \left(\frac{R_{11}}{l_{11}}\right)^{\frac{3}{2}}, \tag{7.107}$$

where $l_{11} = m_{11}^{-1}$ is the Plank length. If we increase the coupling $g_s$, $R_{11}$ grows and the compactified dimension unfolds.

Therefore the $M(2)$-brane tension is

$$T_{M(2)} = \frac{1}{(2\pi)^2} \frac{1}{g_s} \frac{1}{(\alpha')^{\frac{3}{2}}} = T_{D(2)}, \tag{7.108}$$

which means that the $M(2)$-brane is a $D(2)$-brane at strong coupling.

Moreover, in $d = 11$ with a circle-compactified dimension the $M(2)$-brane can wrap $\mathcal{S}_1$ and for a 10-dimensional observer it will look like a string of tension

$$T_{M(2)/\mathcal{S}_1} = 2\pi R T_{M(2)} = \frac{1}{2\pi\alpha'}, \tag{7.109}$$

which is the tension of the fundamental $F(1)$-string. Thus from M-theory we obtained the fundamental objects $D(2)$ and $F(1)$, and also $D(0)$ as the Kaluza-Klein modes of the circle compactification. However there are still $D(4)$, $D(6)$ and $NS(5)$ missing. We can recover them as follows.

The $M(2)$-brane is coupled to the 3-form $A^{(3)}$. If we consider $F^{(4)} = dA^{(3)}$, we can take its Hodge dual and get

$$F^{(7)} = \star_{11} F^{(4)} = dA^{(6)}. \tag{7.110}$$

This $A^{(6)}$ form couples to the "magnetic" counterpart of the $M(2)$-brane, i.e. the $M(5)$-brane, whose tension is naturally given by

$$T_{M(5)} = \frac{m_{11}^6}{(2\pi)^5}. \tag{7.111}$$

This expression is not just natural, but it is dictated by the Dirac quantization condition

$$T_{M(2)} T_{M(5)} = \frac{2\pi}{2K_{11}^2}, \tag{7.112}$$

as one can easily check. In string theory variables, the tension of the $M(5)$-branes becomes the tension of the $NS(5)$-brane

$$T_{M(5)} = \frac{m_{11}^6}{(2\pi)^5} = \frac{1}{(2\pi)^5} \frac{1}{g_s^2} \frac{1}{(\alpha')^3} = T_{NS(5)}. \tag{7.113}$$

Moreover, if we wrap the $M(5)$-brane on the compactified dimension, we get the $D(4)$-brane:

$$T_{M(5)/\mathcal{S}_1} = 2\pi R T_{M(5)} = \frac{1}{(2\pi)^4} \frac{1}{g_s} \frac{1}{(\alpha')^{\frac{5}{2}}} = T_{D(4)}. \tag{7.114}$$

Finally, the $D(6)$-brane can be recovered as a "Kaluza-Klein monopole", the magnetic dual of the KK-momentum modes, which have been identified as the $D(0)$-branes.

All in all, the only parameter of M-theory is $m_{11}$, namely the Plank constant. The theory is therefore inherently strongly coupled and does not have a natural small parameter to do perturbation theory. When we compactify, the radius can be the small parameter and a perturbative window opens up. This perturbative window is in fact the Type IIA superstring. There are other ways to connect M-theory with the five superstring theories and this is summarized in figure 7.1.

## Acknowledgments

We would like to thank Alberto Lerda for useful comments and discussions on a preliminary draft and on related topics. Many thanks to the students who helped in spotting wrong signs, factors and other typos and errors: Davide Bason, Lorenzo de Lillo, Maddalena Ferragatta, Enrico Perron Cabus, Riccardo Poletti, Alberto Ruffino, Emanuele Spirito and Edoardo Vinci.

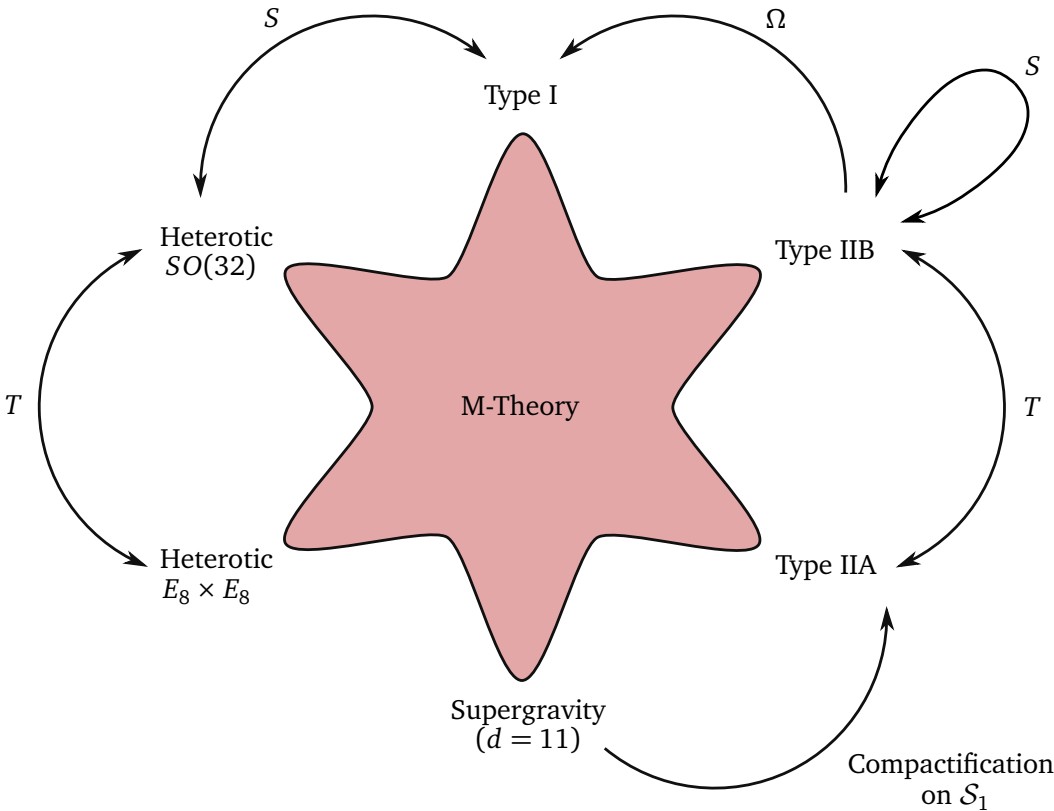

Figure 7.1: M-theory represented as a (yet unknown) land where some patches are the known superstring theories (related to each other by the $T$ and $S$ dualities and the orientifold operation $\Omega$) and the $d = 11$ supergravity, which under a circle compactification reduces to type IIA superstring.

# A Spinors in $d$ (even) dimensions

Let us consider $d$ directions, where $d$ is even. We can group them in pairs that identify mutually orthogonal planes:

$$(0,1),(2,3),\ldots,(d-2,d-1). \tag{A.1}$$

We can label these planes as follows:

$$I=0,1,\ldots,\frac{d}{2}-1=(0,j), \qquad \text{with } j=1,\ldots,\frac{d}{2}-1. \tag{A.2}$$

Consider now the Grassmann odd operators $\psi^\mu$ (with $\mu=0,1,\ldots,d-1$ Lorentz index) which obey the Clifford algebra

$$[\psi^\mu,\psi^\nu]=\eta^{\mu\nu}, \tag{A.3}$$

where as usual $[\cdot,\cdot]$ is the graded commutator. We now define

$$\psi_0^\pm \equiv \frac{i}{\sqrt{2}}\left(\psi_0^0 \pm \psi_0^1\right), \tag{A.4a}$$

$$\psi_j^\pm \equiv \frac{1}{\sqrt{2}}\left(\psi_0^{2j} \pm i\psi_0^{2j+1}\right). \tag{A.4b}$$

We then have that

$$\left[\psi_I^+,\psi_J^-\right]=\delta_{I,J}, \tag{A.5a}$$

$$\left[\psi_I^\pm,\psi_J^\pm\right]=0. \tag{A.5b}$$

---

**Exercise A.1**

Prove eqs. (A.5).

---

Therefore, $\forall I$ we have that $\psi_I^+,\psi_I^-$ generates a two-dimensional space, namely

$$\psi^+\left|+1/2\right\rangle = 0, \tag{A.6a}$$

$$\psi^-\left|+1/2\right\rangle = \left|-1/2\right\rangle, \tag{A.6b}$$

$$\psi^-\left|-1/2\right\rangle = 0, \tag{A.6c}$$

where the basis $\{\left|+1/2\right\rangle,\left|-1/2\right\rangle\}$ generates the two-dimensional space: $\left|+1/2\right\rangle$ is the highest weight state and $\psi^+$ is the raising operator, while $\left|-1/2\right\rangle$ is the lowest weight state and $\psi^-$ is the lowering operator.

Therefore, for a generic even value of $d$, the representation space of the Clifford algebra space has a vacuum of the type

$$\left|0\right\rangle=\underbrace{\left|+1/2\right\rangle\otimes\left|+1/2\right\rangle\otimes\cdots\otimes\left|+1/2\right\rangle}_{\frac{d}{2}\text{ times}}=\underbrace{\left|+1/2,+1/2,\ldots,+1/2\right\rangle}_{\frac{d}{2}\text{ times}}, \tag{A.7}$$

and the generic state is

$$\left|\alpha\right\rangle=\underbrace{\left|\pm1/2\right\rangle\otimes\left|\pm1/2\right\rangle\otimes\cdots\otimes\left|\pm1/2\right\rangle}_{\frac{d}{2}\text{ times}}=\underbrace{\left|\pm1/2,\pm1/2,\ldots,\pm1/2\right\rangle}_{\frac{d}{2}\text{ times}}. \tag{A.8}$$

The label $\alpha$ counts how many of these states one can build. In $d$ (even) dimensions we have that

$$\alpha=1,\ldots,2^{\frac{d}{2}}. \tag{A.9}$$

The state $|\alpha\rangle$ is thus a spinor in $d = 2n$ dimensions (for $n \in \mathbb{N}$), and its related spinor space will have dimension $2^n$.

Let us now consider (recalling eqs. (A.5))

$$\left[\psi_0^+, \psi_0^-\right] = \psi_0^+ \psi_0^- + \psi_0^- \psi_0^+ = \delta_{0,0} = 1 \,. \tag{A.10}$$

Hence we can write

$$-\psi_0^0 \psi_0^1 = \frac{1}{2}\left(\psi_0^+ + \psi_0^-\right)\left(\psi_0^+ - \psi_0^-\right)$$

$$= -\frac{1}{2}\left(\psi_0^- \psi_0^+ - \psi_0^+ \psi_0^-\right)$$

$$= \psi_0^+ \psi_0^- - \frac{1}{2}\,, \tag{A.11a}$$

$$-i\psi_0^{2j} \psi_0^{2j+1} = \psi_j^+ \psi_j^- - \frac{1}{2}\,. \tag{A.11b}$$

We then define

$$\chi_I = \begin{cases} -\psi_0^0 \psi_0^1\,, & \text{if } I = 0, \\ -i\psi_0^{2j} \psi_0^{2j+1}, & \text{if } I = j \end{cases} \quad \Rightarrow \quad \chi_I = \psi_I^+ \psi_I^- - \frac{1}{2}\,, \tag{A.12}$$

and we find that

$$\chi_I \left|+1/2\right\rangle = \psi_I^+ \psi_I^- - \frac{1}{2}\left|+1/2\right\rangle = \left(1 - \frac{1}{2}\right)\left|+1/2\right\rangle = \frac{1}{2}\left|+1/2\right\rangle\,, \tag{A.13a}$$

$$\chi_I \left|-1/2\right\rangle = \psi_I^+ \psi_I^- - \frac{1}{2}\left|-1/2\right\rangle = \left(0 - \frac{1}{2}\right)\left|+1/2\right\rangle = -\frac{1}{2}\left|+1/2\right\rangle\,. \tag{A.13b}$$

This makes clear why we used the notation $\{|+1/2\rangle, |-1/2\rangle\}$ for the basis vectors: we labelled them with their eigenvalues with respect to $\chi_I$.

We can finally define the chirality matrix (which in $d = 4$ is commonly called $\Gamma^5$) as

$$\Gamma = 2^n \prod_{I=0}^{n-1} \chi_I = -(-i)^{n-1} \Gamma^0 \Gamma^1 \ldots \Gamma^{(2n-2)} \Gamma^{(2n-1)}\,, \tag{A.14}$$

where $\Gamma^\mu = \sqrt{2}\psi^\mu$. If we then define

$$|\alpha\rangle = |(\text{even \# of } -1/2)\rangle\,, \tag{A.15a}$$

$$|\dot{\alpha}\rangle = |(\text{odd \# of } -1/2)\rangle\,, \tag{A.15b}$$

the action of the chirality matrix $\Gamma$ on the spinor states is

$$\Gamma |\alpha\rangle = +|\alpha\rangle\,, \tag{A.16a}$$

$$\Gamma |\dot{\alpha}\rangle = -|\dot{\alpha}\rangle\,. \tag{A.16b}$$

We will then say that $|\alpha\rangle$ has a positive chirality, while $|\dot{\alpha}\rangle$ has a negative chirality. They are two irrepses of $SO(1, d-1)$. Indeed the Lorentz group acts on spinors through the matrix

$$\Gamma^{\mu\nu} = [\Gamma^\mu, \Gamma^\nu]\,, \tag{A.17}$$

and since

$$[\Gamma^{\mu\nu}, \Gamma] = 0\,, \tag{A.18}$$

the two chirality representations transform independently of each other under the Lorentz group $SO(1, d-1)$.

# B  Dedekind $\eta$ function

During the computation of the closed string 1-loop vacuum bubble we have defined the Dedekind eta function:

$$\tilde{\eta}(\tau) = q^{\frac{1}{24}} \prod_{l=1}^{\infty} (1 - q^l), \quad \text{with} \quad q = e^{2\pi i \tau}, \quad |q| < 1. \tag{B.1}$$

We will now provide a simple proof of its modular property

$$\eta\left(-\frac{1}{\tau}\right) = \sqrt{-i\tau}\,\eta(\tau). \tag{B.2}$$

If we take the logarithm of $\eta(\tau)$ and expand $\log(1 - q^k)$ in its McLaurin series, convergent in the domain of $\eta$, we get

$$\frac{1}{12} i\pi\tau - \log\eta(\tau) = -\sum_{l=1}^{\infty} \log(1 - q^l) = \sum_{k,l=1}^{\infty} \frac{1}{k} q^{lk} = \sum_{k=1}^{\infty} \frac{1}{k} \frac{1}{q^{-k} - 1}. \tag{B.3}$$

Doing the same computation for $\eta(-1/\tau)$ and using (B.2) we also obtain

$$-\frac{i\pi}{12\tau} - \log\eta\left(-\frac{1}{\tau}\right) = -\frac{i\pi}{12\tau} - \frac{1}{2}\log\frac{\tau}{i} - \log\eta(\tau) = \sum_{k=1}^{\infty} \frac{1}{k} \frac{1}{e^{\frac{2\pi ik}{\tau}} - 1}. \tag{B.4}$$

Then if we take the difference between (B.3) and (B.4), we see that to prove the modular property of $\eta$ it suffices to check that

$$\frac{\pi i}{12}\left(\tau + \frac{1}{\tau}\right) + \frac{1}{2}\log\frac{\tau}{i} = \sum_{k=1}^{\infty} \frac{1}{k}\left[\frac{1}{e^{-2\pi ik\tau} - 1} - \frac{1}{e^{2\pi ik/\tau} - 1}\right]. \tag{B.5}$$

To do so we consider the function $g(z) = \cot(z)\cot(z/\tau)$ and the following meromorphic family:

$$f_\nu(z) = \frac{1}{z} g(\nu z), \quad \nu = \left(n + \frac{1}{2}\right)\pi \qquad (n = 0, 1, \dots). \tag{B.6}$$

The function $f_n u(z)$ has simple poles in $z_k = \pi k/\nu$ and $z_k^\tau = \pm\pi k\tau/\nu$ with residues

$$\text{Res}_{z_k}(f_\nu) = \frac{1}{\pi k}\cot\frac{\pi k}{\tau}, \qquad \text{Res}_{z_k^\tau}(f_\nu) = \frac{1}{\pi k}\cot\pi k\tau, \tag{B.7}$$

for $k \in \mathbb{Z}$. In addition there is a third order pole at $z = 0$ with residue

$$\text{Res}_0(f_\nu) = -\frac{1}{3}\left(\tau + \frac{1}{\tau}\right). \tag{B.8}$$

Let $\mathcal{C}$ be the contour of the rhombus with vertices $1, \tau, 1, -\tau$ as shown in fig. B.1.

Then, integrating over $\mathcal{C}$, by the residues theorem we get

$$\frac{\pi i}{12}\left(\tau + \frac{1}{\tau}\right) + \frac{1}{8}\int_{\mathcal{C}} f_\nu(z)dz = \frac{i}{2}\sum_{k=1}^{n} \frac{1}{k}\left(\cot\pi k\tau + \cot\pi k/\tau\right)$$

$$= \sum_{k=1}^{n} \frac{1}{k}\left[\frac{1}{e^{-2\pi ik\tau} - 1} - \frac{1}{e^{2\pi ik/\tau} - 1}\right]. \tag{B.9}$$

On the other hand, as $n \to \infty$ the function $g(\nu z)$ is uniformly bounded on $\mathcal{C}$ and has on the four sides, excluding the vertices, the limit values $1, -1, 1, -1$. Hence

$$\lim_{n\to\infty} \int_{\mathcal{C}} f_\nu(z)dz = \left(\int_1^\tau - \int_\tau^{-1} + \int_{-1}^{-\tau} - \int_{-\tau}^1\right) \frac{dz}{z} = 4\log\frac{\tau}{i}, \tag{B.10}$$

which gives the desired result.

SciPost Phys. Lect. Notes 90 (2025)

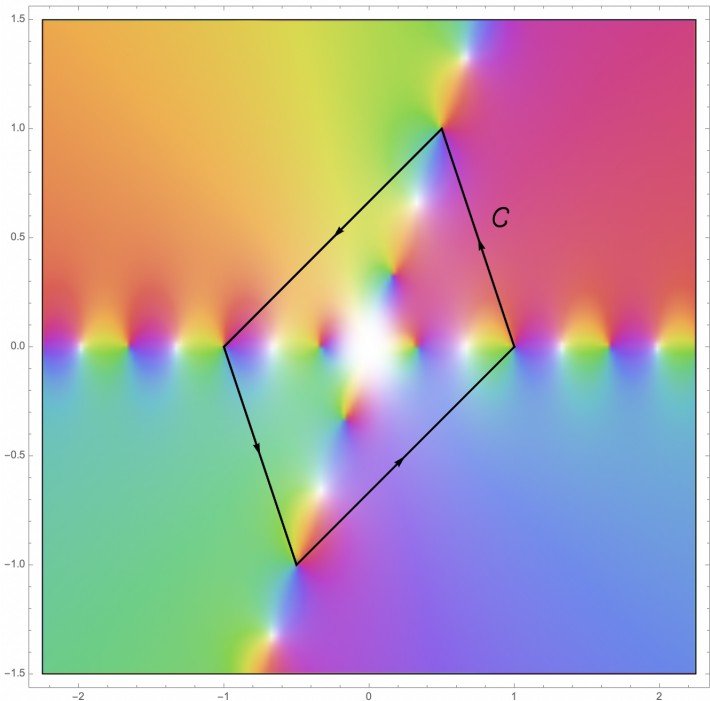

Figure B.1: Plot of $f_\tau(z)$ for $\tau = 1/2 + i$ and $n = 1$. The white spots represent the poles of the meromorphic function. For $n \to \infty$ the number of poles inside the contour $\mathcal{C}$ goes to infinity.

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
