# Peer review of "Introduction to String Theory"

_SciPost Physics Lecture Notes, doi:SciPost Phys. Lect. Notes 90 (2025)_

## Round 2 · Referee Report · Anonymous (Referee 1) · 2024-11-15

Strengths
- The lecture notes are well written and also well organized.
- The lecture notes go beyond the available textbooks and previous lecture notes on string theory.
- The notes contain a complete and very useful discussion of the various quantization methods of string theory, and in this way provide a new perspective on this topic.
Weaknesses
- The emphasis of the lecture notes is mostly on the quantisation of the free bosonic string theory. Other topics, such as the interactions and discussion of the superstring theory, are less well worked out.
- The lecture notes are somewhat "traditional" and do not discuss other important applications of string theory (such as their application black holes and their role in the discovery of the AdS/CFT correspondence).
Report
The authors have written a readable and well-organized Introduction to String Theory in the form of an elaborate set of lecture notes. The notes contain an extensive discussion of various approaches to the quantization of the free string theory for closed and open strings. A strong point is that for the open string that besides the usual Neumann boundary conditions also the Dirichlet boundary conditions are considered, which are necessary for the discussion of D-branes.
There are already quite a few introductory books and online lecture notes on the topic of string theory. Nevertheless, I find that the authors have managed to present this well studied topic in a useful and well-organized manner. In particular, the extensive discussion and comparison of the different quantization methods are not contained in previous expositions of string theory. The lecture notes also discuss the computations of string amplitudes and some aspects of string perturbation theory.
The final chapters describe the extension to the superstring and the role of string dualities. These discussions are less worked out than the earlier chapters on the quantization methods. In this sense the lecture notes are somewhat “traditional” and do not include more recent advances. Topics that are lacking include the applications of string theory and D-branes to the black holes and the AdS/CFT correspondence, topics that are more than 25 years old.
Despite these minor shortcomings, I find that the lecture material is well organized and contains sufficiently many new elements and perspectives to deserve publication in the lecture notes section of SciPost.
There are already quite a few introductory books and online lecture notes on the topic of string theory. Nevertheless, I find that the authors have managed to present this well studied topic in a useful and well-organized manner. In particular, the extensive discussion and comparison of the different quantization methods are not contained in previous expositions of string theory. The lecture notes also discuss the computations of string amplitudes and some aspects of string perturbation theory.
The final chapters describe the extension to the superstring and the role of string dualities. These discussions are less worked out than the earlier chapters on the quantization methods. In this sense the lecture notes are somewhat “traditional” and do not include more recent advances. Topics that are lacking include the applications of string theory and D-branes to the black holes and the AdS/CFT correspondence, topics that are more than 25 years old.
Despite these minor shortcomings, I find that the lecture material is well organized and contains sufficiently many new elements and perspectives to deserve publication in the lecture notes section of SciPost.
Recommendation
Publish (easily meets expectations and criteria for this Journal; among top 50%)

---

## Editorial Decision

published